
# Exact effective interactions and 1/4-BPS dyons in heterotic CHL orbifolds

**Guillaume Bossard[1★], Charles Cosnier-Horeau[1,2] and Boris Pioline[2]**

**1** Centre de Physique Théorique (CPHT), Ecole Polytechnique,
CNRS and Université Paris-Saclay, 91128 Palaiseau Cedex, France
**2** Laboratoire de Physique Théorique et Hautes Energies (LPTHE), CNRS and Sorbonne
Université, Campus Pierre et Marie Curie, 4 place Jussieu, 75005 Paris, France

★ guillaume.bossard@polytechnique.edu

## Abstract

Motivated by precision counting of BPS black holes, we analyze six-derivative couplings in the low energy effective action of three-dimensional string vacua with 16 supercharges. Based on perturbative computations up to two-loop, supersymmetry and duality arguments, we conjecture that the exact coefficient of the $\nabla^2(\nabla\phi)^4$ effective interaction is given by a genus-two modular integral of a Siegel theta series for the non-perturbative Narain lattice times a specific meromorphic Siegel modular form. The latter is familiar from the Dijkgraaf-Verlinde-Verlinde (DVV) conjecture on exact degeneracies of 1/4-BPS dyons. We show that this Ansatz reproduces the known perturbative corrections at weak heterotic coupling, including tree-level, one- and two-loop corrections, plus non-perturbative effects of order $e^{-1/g_3^2}$. We also examine the weak coupling expansions in type I and type II string duals and find agreement with known perturbative results, as well as new predictions for higher genus perturbative contributions. In the limit where a circle in the internal torus decompactifies, our Ansatz predicts the exact $\nabla^2 F^4$ effective interaction in four-dimensional CHL string vacua, along with infinite series of exponentially suppressed corrections of order $e^{-R}$ from Euclideanized BPS black holes winding around the circle, and further suppressed corrections of order $e^{-R^2}$ from Taub-NUT instantons. We show that instanton corrections from 1/4-BPS black holes are precisely weighted by the BPS index predicted from the DVV formula, including the detailed moduli dependence. We also extract two-instanton corrections from pairs of 1/2-BPS black holes, demonstrating consistency with supersymmetry and wall-crossing, and estimate the size of instanton-anti-instanton contributions.



# 1  Introduction

Providing a statistical origin of the thermodynamic entropy of black holes is a key goal for any theory of quantum gravity. More than two decades ago, Strominger and Vafa demonstrated that D-branes of type II string theories provide the correct number of micro-states for supersymmetric black holes in the large charge limit [1]. Since then, much work has gone into performing precise counting of black hole micro-states and comparing with macroscopic supergravity predictions. In vacua with extended supersymmetry, it was found that exact degeneracies of five-dimensional BPS black holes (counted with signs) are given by Fourier coefficients of weak Jacobi forms, giving access to their large charge asymptotics [2–4]. With hindsight, the modular invariance of the partition function of BPS black holes follows from the existence of an $AdS_3$ factor in the near-horizon geometry of these extremal black holes.

In a prescient work [5], Dijkgraaf, Verlinde and Verlinde (DVV) conjectured that four-dimensional BPS black holes in type II string theory compactified on $K3 \times T^2$ (or equivalently, heterotic string on $T^6$) are in fact Fourier coefficients of a meromorphic Siegel modular form, invariant under a larger $Sp(4, \mathbb{Z})$ symmetry. This conjecture was subsequently extended to other four-dimensional vacua with 16 supercharges [6], proven using D-brane techniques [7, 8], and refined to properly incorporate the dependence on the moduli at infinity [9], but the origin of the $Sp(4, \mathbb{Z})$ symmetry had remained obscure. In [10–12], it was noted that a class of 1/4-BPS dyons arises from string networks which lift to M5-branes wrapped on $K3$ times a genus-two curve, but this observation did not yet lead to a transparent derivation of the DVV formula.

In [13], implementing a strategy advocated earlier in [14], we revisited this problem by analyzing certain protected couplings in the low energy effective action of the four-dimensional string theory compactified on a circle of radius $R$ down to three space-time dimensions. In three-dimensional string vacua with 16 or more supercharges, the massless degrees of freedom are described by a non-linear sigma model on a symmetric manifold $G_3/K_3$, which contains the four-dimensional moduli space $\mathcal{M}_4 = G_4/K_4$, the holonomies $a_I^i$ of the four-dimensional gauge fields, the NUT potential $\psi$ dual to the Kaluza–Klein vector and the circle radius $R$. Since stationary solutions with finite energy in dimension 4 yield finite action solutions in dimension 3, it is expected that black holes of mass $\mathcal{M}$ and charge $\Gamma_i^I = (Q^I, P^I)$ in 4 dimensions which break $2r$ supercharges induce instantonic corrections of order $e^{-2\pi R \mathcal{M} + 2\pi i a_i^i \Gamma_i^I}$ to effective couplings with $2r$ fermions (or $r$ derivatives) in dimension 3 (see e.g. [15]); and moreover that these corrections are weighted by the helicity supertrace

$$\Omega_{2r}(\Gamma) = \frac{1}{n!} \mathrm{Tr}_\Gamma[(-1)^F (2h)^n] \,, \tag{1.1}$$

where $F$ is the fermionic parity and $h$ is the helicity in $D = 4$. In addition, there are corrections of order $e^{-2\pi R^2 |M_1| + 2\pi i M_1 \psi}$ from Euclidean Taub-NUT instantons which asymptote to $\mathbb{R}^3 \times S^1$, where the circle is fibered with charge $M_1$ over the two-sphere at spatial infinity. While the two-derivative effective action is uncorrected and invariant under the full continuous group $G_3$, higher-derivative couplings need only be invariant under an arithmetic subgroup $G_3(\mathbb{Z})$ known as the U-duality group. For string vacua with 32 supercharges, the $\mathcal{R}^4, \nabla^4 \mathcal{R}^4$ and $\nabla^6 \mathcal{R}^4$ effective interactions are expected to receive instanton corrections from 1/2-BPS, 1/4-BPS and 1/8-BPS black holes, respectively. In [16], two of the present authors demonstrated that the exact $\mathcal{R}^4, \nabla^4 \mathcal{R}^4$ couplings, given by Eisenstein series for the U-duality group $G_3(\mathbb{Z}) = E_8(\mathbb{Z})$ [17–20], indeed reproduce the respective helicity supertraces $\Omega_4$ and $\Omega_6$ for 1/2-BPS and 1/4-BPS black holes in dimension 4. At the time of writing, a similar check for the $\nabla^6 \mathcal{R}^4$ coupling conjectured in [21] still remains to be performed.

For three-dimensional string vacua with 16 supercharges, the scalar fields span a symmetric space of the form

$$G_3 / K_3 = O(2k, 8) / [O(2k) \times O(8)] \,, \tag{1.2}$$

for a model-dependent integer $k$, which extends the moduli space

$$G_4 / K_4 = SL(2)/SO(2) \times O(2k-2, 6)/[O(2k-2) \times O(6)] \tag{1.3}$$

in $D = 4$. The four-derivative scalar couplings of the form $F(\phi)(\nabla \phi)^4$ are expected to receive instanton corrections from 1/2-BPS black holes, along with Taub-NUT instantons, while six-derivative scalar couplings of the form $G(\phi) \nabla^2 (\nabla \phi)^4$ receive instanton corrections both from four-dimensional 1/2-BPS and 1/4-BPS black holes, along with Taub-NUT instantons. In [13], we restricted for simplicity to the maximal rank case ($k = 12$) arising in heterotic string compactified on $T^7$ (or equivalently type II string theory compactified on $K3 \times T^3$). Using low order perturbative computations, supersymmetric Ward identities and invariance under the U-duality group $G_3(\mathbb{Z}) \subset O(24, 8, \mathbb{Z})$, we determined the tensorial coefficients $F_{abcd}(\phi)$ and $G_{ab,cd}(\phi)$ of the above couplings exactly, for all values of the string coupling. In either case, the non-perturbative coupling is given by a U-duality invariant generalization of the genus-one and genus-two contribution, respectively:

$$F_{abcd}^{(24,8)} \;=\; \text{R.N.} \int_{SL(2,\mathbb{Z}) \backslash \mathcal{H}_1} \frac{d\rho_1 d\rho_2}{\rho_2^2} \frac{\Gamma_{\Lambda_{24,8}}[P_{abcd}]}{\Delta} \,, \tag{1.4}$$

$$G_{ab,cd}^{(24,8)} \;=\; \text{R.N.} \int_{Sp(4,\mathbb{Z}) \backslash \mathcal{H}_2} \frac{d^3\Omega_1 d^3\Omega_2}{|\Omega_2|^3} \frac{\Gamma_{\Lambda_{24,8}}^{(2)}[P_{ab,cd}]}{\Phi_{10}} \,, \tag{1.5}$$

where $\mathcal{H}_h$ is the Siegel upper half-plane of degree $h$, $\Gamma_{\Gamma_{\Lambda_{p,q}}^{(h)}}[P]$ is a genus-$h$ Siegel-Narain theta series for a lattice of signature $(p, q)$ with a polynomial insertion (see (B.4) and (2.32) for the genus-one and two cases), $\Delta$ and $\Phi_{10}$ are the modular discriminants in genus-one and two, and R.N. denotes a specific regularization prescription (see §B.1.3 and §B.2.4 for details). We demonstrated that the Ansätze (1.4) and (1.5) satisfy the relevant supersymmetric Ward identities, and that their asymptotic expansion at weak heterotic string coupling $g_3 \to 0$ reproduces the known perturbative contributions, up to one-loop and two-loop, respectively, plus an infinite series of $\mathcal{O}(e^{-1/g_3^2})$ corrections ascribed to NS5-instantons, Kaluza–Klein (6,1)-branes and H-monopole instantons. We went on to analyze the limit $R \to \infty$ and demonstrated that the $\mathcal{O}(e^{-R})$ corrections to $F_{abcd}^{(24,8)}$ and to $G_{ab,cd}^{(24,8)}$ were proportional to the known helicity supertraces of 1/2- and 1/4-BPS four-dimensional black holes, respectively. In particular, the DVV formula for the index of 1/4-BPS states [5], with the correct contour prescription [9], emerges in a transparent fashion after unfolding the integral over the fundamental domain $Sp(4, \mathbb{Z}) \backslash \mathcal{H}_2$ onto the full Siegel upper-half plane $\mathcal{H}_2$.

In [22], we extended the study of the 1/2-BPS saturated coupling $F_{abcd}^{(24,8)}$ to the case of CHL heterotic orbifolds of prime order $N = 2, 3, 5, 7$. All these models have 16 supercharges, and their moduli space in $D = 3$ or 4 is of the form (1.2), (1.3) with $k = 24/(N+1)$ [25]. After conjecturing the precise form of the U-duality group $G_3(\mathbb{Z}) \subset O(2k, 8, \mathbb{Z})$ in $D = 3$, we proposed an exact formula for $F_{abcd}$ analogous to (1.4),

$$F_{abcd}^{(2k,8)} = \text{R.N.} \int_{\Gamma_0(N) \backslash \mathcal{H}_1} \frac{\mathrm{d}\rho_1 \mathrm{d}\rho_2}{\rho_2^2} \frac{\Gamma_{\Lambda_{2k,8}}[P_{abcd}]}{\Delta_k} \, , \tag{1.6}$$

where $\Gamma_0(N) \subset SL(2, \mathbb{Z})$ is the Hecke congruence subgroup of level $N$, $\Delta_k$ is the unique cusp form of weight $k$ under $\Gamma_0(N)$ and $\Lambda_{2k,8}$ is the 'non-perturbative Narain lattice' of signature $(2k, 8)$. We studied the weak coupling and large radius limits of the Ansatz (1.6), and found that it reproduces correctly the known tree-level and one-loop contributions in the limit $g_3 \to 0$, powerlike corrections in the limit $R \to \infty$, as well as infinite series of instanton corrections consistent with the known helicity supertrace of 1/2-BPS states in $D = 4$, for all orbits of the U-duality group $G_4(\mathbb{Z})$.

The goal of the present work is to provide strong evidence that the tensorial coefficient $G_{ab,cd}$ of the 1/4-BPS saturated coupling $\nabla^2(\nabla\phi)^4$ in the same class of CHL orbifolds is given by the natural extension of (1.5), namely

$$G_{ab,cd}^{(2k,8)} = \text{R.N.} \int_{\Gamma_{2,0}(N) \backslash \mathcal{H}_2} \frac{\mathrm{d}^3\Omega_1 \mathrm{d}^3\Omega_2}{|\Omega_2|^3} \frac{\Gamma_{\Lambda_{2k,8}}^{(2)}[P_{ab,cd}]}{\Phi_{k-2}} \, , \tag{1.7}$$

where $\Gamma_{2,0}(N)$ is a congruence subgroup of level $N$ inside $Sp(4, \mathbb{Z})$, $\Phi_{k-2}$ is a specific meromorphic Siegel modular form of weight $k-2$ and $\Gamma_{\Lambda_{2k,8}}^{(2)}[P_{ab,cd}]$ is a suitable genus-two Siegel-Narain theta series for the same non-perturbative Narain lattice $\Lambda_{2k,8}$ as in (1.6). Using similar techniques as in [22], we find that

- the Ansatz (1.7) satisfies the relevant supersymmetric Ward identities and produces the correct tree-level, one-loop and two-loop terms in the weak heterotic coupling limit;

- in the decompactification limit, the Ansatz predicts the exact $\nabla^2 F^4$ and $F^2\mathcal{R}^2$ couplings in 4 dimensions, extending known perturbative and non-perturbative results in type I and type II string vacua with reduced rank;

- the effective coupling provides a duality-invariant generating function for the indices (or helicity supertraces) counting 1/4-BPS black hole micro-states in 4 dimensions, which arise as coefficients of exponentially suppressed contributions in the large radius limit;

- in particular, we verify the standard formula for the index of 1/4-BPS black holes with "simple" primitive charges and obtain a prediction for all possible charge orbits.

From a more technical point of view, a significant feature and complication of (1.5),(1.7) compared to the 1/2-BPS coupling (1.4), (1.6) is that the integrand $1/\Phi_{k-2}$ has a double pole on the diagonal locus $\Omega_{12} = 0$ and its images under $\Gamma_{2,0}(N)$ (corresponding to the separating degeneration of the genus-two Riemann surface with period matrix $\Omega$). In the context of the DVV formula, these poles are well-known to be responsible for the moduli dependence of the helicity supertrace $\Omega_6$. In the context of the BPS coupling (1.7), these poles are responsible for the fact that the weak coupling and large radius expansions receive infinite series of instanton anti-instanton contributions, as required by the quadratic source term in the differential equation (2.26) for the coefficient $G_{ab,cd}^{(2k,8)}$. A similar phenomenon has encountered in the case of the $\nabla^6\mathcal{R}^4$ couplings in maximal supersymmetric string vacua [23].

## Organization

This work is organized as follows. In §2 we recall relevant facts about the moduli space, duality group and BPS spectrum of heterotic CHL models in $D = 4$ and $D = 3$, record the known perturbative contributions to the $\nabla^2 F^4$ and $\nabla^2 (\nabla \phi)^4$ couplings in heterotic perturbation theory, and summarize our main results. The remainder of the text provides extensive details on the derivation of these results, the hurried reader may skip §3 and proceed to the discussions in §4.3, §5.3 and §6 of the consequences of our Ansatz for the exact $\nabla^2 F^4$ and $\nabla^2 (\nabla \phi)^4$ couplings.

In §3, we derive the differential constraints imposed by supersymmetry on these couplings, and show that they are obeyed by the Ansatz (1.7). In §4, we study the expansion of (1.7) at weak heterotic coupling, and show that it correctly reproduces the known pertubative contributions, along with an infinite series of NS5-brane, Kaluza–Klein (6,1)-branes and H-monopole instanton corrections. In §5, we move to the central topic of this work and study the large radius limit of the Ansatz (1.7). We obtain the exact $\nabla^2 F^4$ and $\mathcal{R}^2 F^2$ couplings in $D = 4$ plus infinite series of $\mathcal{O}(e^{-R})$ and $\mathcal{O}(e^{-R^2})$ corrections. We extract from the former the helicity supertrace of 1/4-BPS black holes with arbitrary charge, and recover the DVV formula and its generalizations. We further analyze two-instanton contributions from pairs of 1/2-BPS black holes and show their consistency with wall-crossing. In §6 we study the weak coupling limit of the $\nabla^2 (\nabla \phi)^4$ couplings in CHL orbifolds of type II string on $K3 \times T^3$ and type I string on $T^7$, and of the related $\nabla^2 \mathcal{H}^4$ couplings in type IIB compactified on K3 down to six dimensions.

A number of more technical developments are relegated to appendices. In Appendix A we collect relevant facts about genus-two Siegel modular forms, and the structure of their Fourier and Fourier-Jacobi expansions. In §B we compute the one-loop and two-loop contributions to the $\nabla^2 F^4$ and $\nabla^2 (\nabla \phi)^4$ couplings in CHL models, spell out the regularization of the corresponding modular integrals, compute the anomalous terms in the differential constraints due to boundary contributions, and discuss their behavior near points of enhanced gauge symmetry. In §C, we verify that the polar contributions to the Fourier coefficients of $1/\Phi_{k-2}$ are in one-to-one correspondence with the possible splittings $\Gamma = \Gamma_1 + \Gamma_2$ of a 1/4-BPS charge $\Gamma$ into a pair of 1/2-BPS charges $\Gamma_1, \Gamma_2$. In §D, we use this information to compute the singular contributions to Abelian Fourier coefficients with generic 1/4-BPS charge, and in §E demonstrate that the structure of these coefficients and of the constant terms is consistent with the differential constraint. In §F, we also estimate the corrections to the saddle point value of the Abelian Fourier coefficients, due to the non-constancy of the Fourier coefficients of $1/\Phi_{k-2}$ and show that they are of the size expected for two-instanton effects on the one hand, and Taub-NUT instanton – anti-instanton on the other hand. In §G, we explain how to infer the non-Abelian Fourier coefficients with respect to $O(p-2, q-2)$ from the knowledge of the Abelian coefficients with respect to $O(p-1, q-1)$. Finally, §H collects definitions of various polynomials which enter in the formulae of §4 and §5.1.

**Note:** The structure of the body of this paper follows that of our previous work [22] on 1/2-BPS couplings, so as to facilitate comparison between our treatments of the genus-one and genus-two modular integrals. The reader is invited to refer to [22] for more details on points discussed cursorily herein.

## 2 Background and executive summary

In this section, we recall relevant facts about the moduli space, duality group and BPS spectrum of heterotic CHL models in $D = 4$ and $D = 3$, and summarize the main features of our Ansatz

for the exact $\nabla^2(\nabla\phi)^4$ and $\nabla^2 F^4$ couplings in these models. Further discussion of the physical consequences of our Ansatz is deferred to Sections §4.3, §5.3 and §6, after the details of the derivation of the results previewed in this section have been provided.

## 2.1 Moduli spaces and dualities

Recall that heterotic CHL models are freely acting orbifolds of the heterotic string compactified on a torus, preserving 16 supersymmetries [24, 25] (see [26] for a review of these constructions). We shall be mostly interested in models with $D = 4$ non-compact spacetime dimensions, and the reduction of those models on a circle down to $D = 3$. Furthermore, for simplicity we restrict to $\mathbb{Z}_N$ orbifolds with $N \in \{1, 2, 3, 5, 7\}$ prime, in which case the gauge symmetry in $D = 4$ is reduced from $U(1)^{28}$ in the original 'maximal rank' model (namely, heterotic string compactified on $T^6$) to $U(1)^{2k+4}$ with $k = 24/(N+1)$. The lattice of electromagnetic charges in $D = 4$ is a direct sum $\Lambda_{em} = \Lambda_e \oplus \Lambda_m$ where $\Lambda_m$ is an even, lattice of signature $(2k-2, 6)$ and $\Lambda_e \equiv \Lambda_m^*$ its dual (see Table 1 in [22]). While $\Lambda_m$ is not self-dual for $N > 1$, it is $N$-modular in the sense that $\Lambda_m^* = \Lambda_m[1/N]$ [27], in particular we have the chain of inclusions $N\Lambda_m \subset N\Lambda_m^* \subset \Lambda_m \subset \Lambda_m^*$.

The moduli space in $D = 4$ is a quotient

$$\mathcal{M}_4 = G_4(\mathbb{Z})\backslash \left[ SL(2,\mathbb{R})/SO(2) \times G_{2k-2,6} \right] , \tag{2.1}$$

where $SL(2,\mathbb{R})/SO(2)$ is parametrized by the heterotic axiodilaton $S$ and the Grassmannian $G_{2k-2,6}$ is parametrized by the scalars $\varphi$ in the vector multiplets. Here and elsewere, $G_{p,q}$ will denote the orthogonal Grassmannian $O(p,q)/[O(p) \times O(q)]$ of negative $q$-planes in $\mathbb{R}^{p,q}$. The U-duality group $G_4(\mathbb{Z})$ in $D = 4$ includes the S-duality group $\Gamma_1(N)$ acting on the first factor and the T-duality group $\widetilde{O}(2k-2, 6, \mathbb{Z})$ acting on the second (where $\widetilde{O}(2k-2, 6, \mathbb{Z})$ denotes the restricted automorphism group of $\Lambda_m$, acting trivially on the discriminant group $\Lambda_e/\Lambda_m \sim \mathbb{Z}_N^{k+2}$). As discussed in [22, 27], there are strong indications that BPS observables are invariant under the larger group $\Gamma_0(N) \times O(2k-2, 6, \mathbb{Z})$, the automorphism group of $\Lambda_m$, along with the Fricke involution acting by $S \mapsto -1/(NS)$ on the first factor, accompanied by a suitable action $\varphi \mapsto \varsigma \cdot \varphi$ of $\varsigma \in O(2k-2, 6, \mathbb{R})$ on the second factor, such that $\Lambda_m^* = \varsigma \cdot \Lambda_m/\sqrt{N}$.

After compactification on a circle of radius $R$ down to $D = 3$, the moduli space spanned by the scalars $\phi = (R, S, \varphi, a^{Ii}, \psi)$ described in the introduction becomes a quotient

$$\mathcal{M}_3 = G_3(\mathbb{Z})\backslash G_{2k,8} \tag{2.2}$$

of the orthogonal Grassmannian $G_{2k,8}$ by the U-duality group in $D = 3$. In [22], generalizing [28] we provided evidence that the U-duality group includes the restricted automorphism group $\widetilde{O}(2k, 8, \mathbb{Z})$ of the 'non-perturbative Narain lattice' [1]

$$\Lambda_{2k,8} = \Lambda_m \oplus I\!I_{1,1} \oplus I\!I_{1,1}[N] , \tag{2.3}$$

which is also $N$-modular. It also includes the U-duality group $G_4(\mathbb{Z})$ as well as the restricted automorphism group $\widetilde{O}(2k-1, 7, \mathbb{Z})$ of the perturbative Narain lattice $\Lambda_m \oplus I\!I_{1,1}$. The exact four and six-derivative couplings of interest in this paper will turn out to be invariant under

---

[1] Note that the non-perturbative Narain lattice' determines the U-duality group, but it does not define a lattice of non-perturbative charges that would complete the perturbative charge lattice. In three dimensions, the analogue of particle charges are elements of the U-duality group $G_3(\mathbb{Z})$ measuring the monodromy of scalar fields around the point particle. Similarly the modular integrals (1.4) and (1.5) should not be interpreted as some putative 'non-perturbative string amplitude', but rather as mathematical functions that turn out to have all the required properties to represent the exact non-perturbative couplings. In particular, the supersymmetry Ward identities satisfied by these threshold functions are essential in ensuring equality with these special automorphic functions.

the full automorphism group $O(2k, 8, \mathbb{Z}) \supset G_3(\mathbb{Z})$, however, this group is not expected to be a symmetry of the full spectrum. In particular, the automorphism group of the perturbative lattice $O(2k-1, 7, \mathbb{Z})$ does not preserve the orbifold projection, and does not act consistently on states that are not invariant under the $\mathbb{Z}_N$ action on the circle. Nevertheless, we expect the U-duality group to be larger than $\widetilde{O}(2k, 8, \mathbb{Z})$ and to include in particular Fricke duality.

An important consequence of the enhancement of T-duality group $\widetilde{O}(2k-1, 7, \mathbb{Z})$ to the U-duality group $\widetilde{O}(2k, 8, \mathbb{Z})$ is that singularities in the low energy effective action occur on codimension-8 loci in the full moduli space $\mathcal{M}_3$, partially resolving the singularities which occur at each order in the perturbative expansion on codimension-7 loci where the gauge symmetry is enhanced.

## 2.2 BPS dyons in $D = 4$

We now review relevant facts about helicity supertraces of 1/2-BPS and 1/4-BPS states in heterotic CHL orbifolds. As mentioned above, the lattice of electromagnetic charges $\Gamma = (Q, P)$ decomposes into $\Lambda_{em} = \Lambda_m^* \oplus \Lambda_m$, where on the heterotic side the first factor corresponds to electric charges $Q$ carried by fundamental heterotic strings, while the second factor corresponds to magnetic charges $P$ carried by heterotic five-brane, Kaluza-Klein (6,1)-brane and H-monopoles. The lattices $\Lambda_e = \Lambda_m^*$ and $\Lambda_m$ carry quadratic forms such that

$$Q^2 \in \frac{2}{N}\mathbb{Z}\,, \quad P^2 \in 2\mathbb{Z}, \quad P \cdot Q \in \mathbb{Z}\,, \tag{2.4}$$

while $\Lambda_{em}$ carries the symplectic Dirac pairing $\langle \Gamma, \Gamma' \rangle = Q \cdot P' - Q' \cdot P \in \mathbb{Z}$. A generic BPS state with charge $\Gamma \in \Lambda_{em}$ such that $Q \wedge P \neq 0$ (i.e. when $Q$ and $P$ are not collinear) preserves 1/4 of the 16 supercharges, and has mass

$$\mathcal{M}(\Gamma; t) = \sqrt{2\frac{|Q_R + SP_R|^2}{S_2} + 4\sqrt{\left| \begin{matrix} Q_R^2 & Q_R \cdot P_R \\ Q_R \cdot P_R & P_R^2 \end{matrix} \right|}}, \tag{2.5}$$

where $t = (S, \varphi)$ denote the set of all coordinates on (2.1), and $Q_R, P_R$ are the projections of the charges $Q, P$ on the negative 6-plane parametrized by $\varphi \in G_{2k-2,6}$. When $Q \wedge P = 0$, the state preserves half of the 16 supercharges, and the mass formula (2.5) reduces to $\mathcal{M}(\Gamma)^2 = 2|Q_R + SP_R|^2/S_2$.

In order to describe the helicity traces carried by these states, it is useful to distinguish 'untwisted' 1/2-BPS states, characterized by the fact that their charge vector $(Q, P)$ lies in the sublattice $\Lambda_m \oplus N\Lambda_e \subset \Lambda_e \oplus \Lambda_m$, from 'twisted' 1/2-BPS states where $(Q, P)$ lies in the complement of this sublattice inside $\Lambda_{em}$. One can show that twisted 1/2-BPS states lie in two different orbits of the S-duality group $\Gamma_0(N)$: they are either dual to a purely electric state of charge $(Q, 0)$ with $Q \in \Lambda_e \smallsetminus \Lambda_m$, or to a purely magnetic state of charge $(0, P)$ with $P \in \Lambda_m \smallsetminus N\Lambda_e$. Similarly, untwisted 1/2-BPS states are either dual to a purely electric state of charge $(Q, 0)$ with $Q \in \Lambda_m$, or to a purely magnetic state of charge $(0, P)$ with $P \in N\Lambda_e$. The fourth helicity supertrace is sensitive to 1/2-BPS states only, and is given by

$$\Omega_4(\Gamma) = c_k\left(-\frac{\gcd(NQ^2, P^2, Q \cdot P)}{2}\right) \tag{2.6}$$

for twisted electromagnetic charge $\Gamma \in (\Lambda_e \oplus \Lambda_m) \smallsetminus (\Lambda_m \oplus N\Lambda_e)$, and by

$$\Omega_4(\Gamma) = c_k\left(-\frac{\gcd(NQ^2, P^2, Q \cdot P)}{2}\right) + c_k\left(-\frac{\gcd(NQ^2, P^2, Q \cdot P)}{2N}\right) \tag{2.7}$$

for untwisted charge $\Gamma \in \Lambda_m \oplus N\Lambda_e$. Here, the $c_k$'s are the Fourier coefficient of $1/\Delta_k = \sum_{m \geq 1} c_k(m)q^m = \frac{1}{q} + k + \dots$, where $\Delta_k = \eta^k(\tau)\eta^k(N\tau)$ is the unique cusp form of weight $k$ under $\Gamma_0(N)$. In the maximal rank case $N = 1$, we write $c(m) = c_{12}(m)$ for brevity.

In contrast, the helicity supertrace $\Omega_6$ is sensitive to 1/2-BPS and 1/4-BPS states. For 1/4-BPS charge $Q \wedge P \neq 0$, it is given by a Fourier coefficient of a meromorphic Siegel modular form [5, 6, 8]. In the simplest instance corresponding to dyons with 'generic twisted' charge $\Gamma = (Q, P)$ in $(\Lambda_e \smallsetminus \Lambda_m) \oplus (\Lambda_m \smallsetminus N\Lambda_e)$ and unit 'torsion' $(\gcd(Q \wedge P) = 1)$, [2]

$$\Omega_6(\Gamma; t) = \frac{(-1)^{Q \cdot P + 1}}{N} \int_{\mathcal{C}} d^3\Omega \, \frac{e^{i\pi[\rho Q^2 + \sigma P^2 + 2\nu Q \cdot P]}}{\tilde{\Phi}_{k-2}(\rho, \sigma, \nu)} \, , \tag{2.8}$$

where the contour $\mathcal{C}$ in the Siegel upper half plane $\mathcal{H}_2$ parametrized by $\Omega = \begin{pmatrix} \rho & \nu \\ \nu & \sigma \end{pmatrix}$ is given by $0 \leq \rho_1 \leq N$, $0 \leq \sigma_1 \leq 1$, $0 \leq \nu_1 \leq 1$ with a fixed value $\Omega_2$ of $(\rho_2, \sigma_2, \nu_2)$ (see below). The overall sign $(-1)^{Q \cdot P + 1}$ ensures that contributions from single-centered black holes are positive [31, 32]. Here, $\tilde{\Phi}_{k-2}(\rho, \sigma, \nu)$ is a Siegel modular form of weight $k - 2$ under a congruence subgroup $\tilde{\Gamma}_{2,0}(N) \subset Sp(4, \mathbb{Z})$ which is conjugate to the standard Hecke congruence subgroup

$$\Gamma_{2,0}(N) = \left\{ \begin{pmatrix} A & B \\ C & D \end{pmatrix} \subset Sp(4, \mathbb{Z}), C = 0 \bmod N \right\} \tag{2.9}$$

by the transformation $S_\rho$ defined in (A.10). $\tilde{\Phi}_{k-2}(\rho, \sigma, \nu)$ is the image of a Siegel modular form $\Phi_{k-2}(\rho, \sigma, \nu)$ of weight $k - 2$ under $\Gamma_{2,0}(N)$ under the same transformation,

$$\tilde{\Phi}_{k-2}(\rho, \sigma, \nu) = (\sqrt{N})^k (-i\rho)^{-(k-2)} \Phi_{k-2}\left( -\frac{1}{\rho}, \sigma - \frac{\nu^2}{\rho}, \frac{\nu}{\rho} \right). \tag{2.10}$$

Ignoring the choice of contour $\mathcal{C}$, (2.8) is formally invariant under the U-duality group $\Gamma_0(N) \times O(2k - 2, 6, \mathbb{Z})$, the first factor acting as the block diagonal subgroup (A.14) of the congruence subgroup $\tilde{\Gamma}_{2,0}(N)$. Invariance under Fricke S-duality follows from the invariance of $\tilde{\Phi}_{k-2}$ under the genus-two Fricke involution (A.39). Note that the sign $(-1)^{Q \cdot P + 1}$ also is invariant under $\Gamma_0(N) \times O(2k - 2, 6, \mathbb{Z})$ and Fricke S-duality.

Importantly, both $\Phi_{k-2}(\rho, \sigma, \nu)$ and $\tilde{\Phi}_{k-2}(\rho, \sigma, \nu)$ have a double zero on the diagonal divisor $\mathcal{D}$ given by all images of the locus $\nu = 0$ under $\Gamma_{2,0}(N)$. Hence, the right-hand side of (2.8) jumps whenever the contour $\mathcal{C}$ crosses $\mathcal{D}$. As explained in [9, 33], if one chooses the constant part of $\Omega_2$ along the contour $\mathcal{C}$ in terms of the moduli $t$ and charge $\Gamma$ via

$$\Omega_2^* = \frac{R}{\mathcal{M}(Q, P)} \left[ \frac{1}{S_2} \begin{pmatrix} 1 & S_1 \\ S_1 & |S|^2 \end{pmatrix} + \frac{1}{|Q_R \wedge P_R|} \begin{pmatrix} P_R^2 & -Q_R \cdot P_R \\ -Q_R \cdot P_R & Q_R^2 \end{pmatrix} \right], \tag{2.11}$$

where $R$ is a large positive number (identified in our set-up as the radius of the circle), then $\mathcal{C}$ crosses $\mathcal{D}$ precisely when the moduli allow for the marginal decay of the 1/4-BPS state of charge $\Gamma = \Gamma_1 + \Gamma_2$ into a pair of 1/2-BPS states with charges $\Gamma_1$ and $\Gamma_2$. The corresponding jump in $\Omega_6(Q, P; t)$ can then be shown to agree [34–36] with the primitive wall-crossing formula [37]

$$\Delta\Omega_6(\Gamma) = -(-1)^{\langle \Gamma_1, \Gamma_2 \rangle + 1} \Omega_4(\Gamma_1) \Omega_4(\Gamma_2) \, , \tag{2.12}$$

where $\Delta\Omega_6$ is the index in the chamber where the bound state exists, minus the index in the chamber where it does not.

The formula (2.8) only applies to dyons whose charge is primitive with unit torsion and that is generic, in the sense that it belongs to the highest stratum in the following graph of

---

[2] Using (A.39), one may rewrite (2.8) in the other common form (see e.g. [29, (5.1.10)])

$$\Omega_6(\Gamma; t) = \frac{(-1)^{Q \cdot P + 1}}{N} \int_{\mathcal{C}'} d^3\Omega \, \frac{e^{i\pi[N\rho Q^2 + \sigma P^2/N + 2\nu Q \cdot P]}}{\tilde{\Phi}_{k-2}(\sigma, \rho, \nu)} \, .$$

Note that our $\tilde{\Phi}_{k-2}$ differs from the one in [29] by an exchange of $\rho$ and $\sigma$, but agrees with $\Phi_{g,e}(\rho, \sigma, \nu)$ in [30].

inclusions [3]

$$
\begin{matrix}
N\Lambda_e \oplus N\Lambda_e \\
\Lambda_m \oplus N\Lambda_m
\end{matrix}
\begin{matrix} \subset \\ \subset \end{matrix}
\Lambda_m \oplus N\Lambda_e
\begin{matrix} \subset \\ \subset \end{matrix}
\begin{matrix}
\Lambda_m \oplus \Lambda_m \\
\Lambda_e \oplus N\Lambda_e
\end{matrix}
\begin{matrix} \subset \\ \subset \end{matrix}
\Lambda_e \oplus \Lambda_m \,.
\tag{2.13}
$$

When $(Q,P)$ is primitive and belongs to one of the sublattices above, it may split into pairs of 1/2-BPS charges that are not necessarily 'twisted' nor primitive. As explained in [13], the study of 1/4-BPS couplings in $D = 3$ provides a microscopic motivation for the contour prescription (2.11), and gives access to the helicity supertrace for arbitrary charges in (2.13) beyond the special case of the highest stratum for which (2.8) is valid.

Indeed, it will follow from the analysis in the present work that for any primitive charge $(Q,P)$, the helicity supertrace is given by

$$
\begin{aligned}
(-1)^{Q\cdot P+1}\Omega_6(Q,P;t) = {}& \sum_{\substack{A\in M_{2,0}(N)/(\mathbb{Z}_2 \rtimes \Gamma_0(N)) \\ A^{-1}\binom{Q}{P}\in\Lambda_e\oplus\Lambda_m}} |A|\, \widetilde{C}_{k-2}\left[A^{-1}\begin{pmatrix} -Q^2 & -Q\cdot P \\ -Q\cdot P & -P^2 \end{pmatrix}A^{-1\mathsf{T}}; A\Omega_2^\star A^\mathsf{T}\right] \\
& + \sum_{\substack{A\in M_2(\mathbb{Z})/GL(2,\mathbb{Z}) \\ A^{-1}\binom{Q}{P}\in\Lambda_m\oplus\Lambda_m}} |A|\, C_{k-2}\left[A^{-1}\begin{pmatrix} -Q^2 & -Q\cdot P \\ -Q\cdot P & -P^2 \end{pmatrix}A^{-1\mathsf{T}}; A\Omega_2^\star A^\mathsf{T}\right] \\
& + \sum_{\substack{A\in M_2(\mathbb{Z})/GL(2,\mathbb{Z}) \\ A^{-1}\binom{Q}{P/N}\in\Lambda_e\oplus\Lambda_e}} |A|\, C_{k-2}\left[A^{-1}\begin{pmatrix} -NQ^2 & -Q\cdot P \\ -Q\cdot P & -\frac{1}{N}P^2 \end{pmatrix}A^{-1\mathsf{T}}; A\Omega_2^\star A^\mathsf{T}\right],
\end{aligned}
\tag{2.14}
$$

where $C_{k-2}$ and $\widetilde{C}_{k-2}$ are the Fourier coefficients of $1/\Phi_{k-2}$ and $1/\tilde{\Phi}_{k-2}$ evaluated with the same contour prescription as above, and $\Omega_2^\star$ is conjugated by the matrix $A$. This formula is manifestly invariant under the U-duality group $G_4(\mathbb{Z})$, including Fricke duality that exchanges the last two lines. For primitive 'twisted charges' of $\gcd(Q\wedge P) = 1$, only the first line is non-zero and the only allowed matrix $A$ is the identity such that one recovers (2.8). This is also the dominant term in the limit where the charges $Q, P$ are scaled to infinity, since terms with $A\neq 1$ in the sum grow exponentially as $e^{\pi|Q\wedge P|/|\det A|}$, at a much slower rate that the leading term with $A = 1$ [2,8]. It would be interesting to check that the logarithmic corrections to the black hole entropy are consistent with the $\mathcal{R}^2$ coupling in the low energy effective action, generalizing the analysis of [6,38] to general charges, and to identify the near horizon geometries responsible for the exponentially suppressed contributions, along the lines of [39,40].

After splitting $C_{k-2}$ and $\widetilde{C}_{k-2}$ into their finite and polar parts, and representing the latter as a Poincaré sum, we shall show that the unfolded sum over matrices $A$ accounts for all possible splittings of a charge $(Q,P) = (Q_1,P_1) + (Q_2,P_2)$ into two 1/2-BPS constituents, labeled by $A \sim \begin{pmatrix} p & q \\ r & s \end{pmatrix} \in M_2(\mathbb{Z})$ [34],

$$
(Q_1,P_1) = (p,r)\frac{sQ - qP}{ps - qr}, \qquad (Q_2,P_2) = (q,s)\frac{pP - rQ}{ps - qr}.
\tag{2.15}
$$

Generalizing the analysis in [41], we shall show that the discontinuity of $\Omega_6(\Gamma, t)$ for an arbitrary primitive (but possibly torsionful) charge $\Gamma$ is given by a variant of (2.12) where $\Omega_4(\Gamma)$ on the right-hand side is replaced by

$$
\overline{\Omega}_4(\Gamma) = \sum_{\substack{d\geq 1 \\ \Gamma/d \in \Lambda_e\oplus\Lambda_m}} \Omega_4(\Gamma/d),
\tag{2.16}
$$

in agreement with the macroscopic analysis in [36].

---

[3]The graph is drawn such that Fricke duality acts by reflection with respect to the horizontal axis.

## 2.3 BPS couplings in $D = 4$ and $D = 3$

In supersymmetric string vacua with 16 supercharges, the low-energy effective action at two-derivative order is exact at tree level, being completely determined by supersymmetry. In contrast, four-derivative and six-derivative couplings may receive quantum corrections from 1/2-BPS and 1/4-BPS states or instantons, respectively. At four-derivative order, the coefficients of the $\mathcal{R}^2 + F^4$ and $F^4$ couplings in $D = 4$, which we denote by $f$ and $F_{abcd}^{(2k-2,6)}$, are known exactly, and depend only on the first and second factor in the moduli space (2.1), respectively:

$$f(S) = -\frac{3}{8\pi^2} \log(S_2^k |\Delta_k(S)|^2), \qquad (2.17)$$

$$F_{abcd}^{(2k-2,6)} = \text{R.N.} \int_{\Gamma_0(N)\backslash\mathcal{H}_1} \frac{d\rho_1 d\rho_2}{\rho_2^2} \frac{\Gamma_{\Lambda_{2k-2,6}}[P_{abcd}]}{\Delta_k(\rho)}, \qquad (2.18)$$

where $\Gamma_{\Lambda_{2k-2,6}}[P_{abcd}]$ denotes the Siegel–Narain theta series (B.4) for the lattice $\Lambda_m = \Lambda_{2k-2,6}$, with an insertion of the symmetric polynomial

$$P_{abcd}(Q) = Q_{L,a} Q_{L,b} Q_{L,c} Q_{L,d} - \frac{3}{2\pi\rho_2} \delta_{(ab} Q_{L,c} Q_{L,d)} + \frac{3}{16\pi^2 \rho_2^2} \delta_{(ab} \delta_{cd)}, \qquad (2.19)$$

and $\Delta_k$ is the same cusp form whose Fourier coefficients enter in the helicity supertrace (2.6),(2.7). Here and elsewhere, we suppress the dependence of $\Gamma_{\Lambda_{p,q}}[P_{abcd}]$, and therefore of the left-hand side of (2.18), on the moduli $\phi \in G_{p,q}$. The regularization prescription used in defining (2.18) is detailed in §B.1.3. As explained in [22], both couplings arise as polynomial terms in the large radius limit of the exact $(\nabla\phi)^4$ coupling in $D = 3$. The latter is uniquely determined by supersymmetry Ward identities, invariance under U-duality and the tree-level and one-loop corrections in heterotic perturbation string theory to be given by the genus-one modular integral (1.4). In the weak heterotic coupling limit $g_3 \to 0$, (1.4) has an asymptotic expansion

$$g_3^2 F_{abcd}^{(2k,8)} = \frac{3}{2\pi g_3^2} \delta_{(ab} \delta_{cd)} + F_{abcd}^{(2k-1,7)} + \sum_{Q \in \Lambda_{2k-1,7}}' \bar{c}_k(Q) e^{-\frac{2\pi\sqrt{2}|Q_R|}{g_3^2} + 2\pi i a \cdot Q} \mathcal{P}_{abcd}^{(*)}, \qquad (2.20)$$

reproducing the known tree-level and one-loop corrections, along with an infinite series of $\mathcal{O}(e^{-1/g_3^2})$ corrections ascribable to NS5-brane, Kaluza–Klein (6,1)-brane and H-monopole instantons. Here, $\mathcal{P}_{abcd}^{(*)}$ is a schematic notation for the tensor appearing in front of the exponential, including an infinite series of subleading terms which resum into a Bessel function. In the large radius limit $R \to \infty$, the asymptotic expansion of (1.4) instead gives, schematically,

$$\begin{aligned}
F_{abcd}^{(2k,8)}(\phi) =\ & R^2 \left( f(S) \delta_{(ab} \delta_{cd)} + F_{abcd}^{(2k-2,6)}(\varphi) \right) \\
& + \sum_{\substack{(Q,P) \in \Lambda_e \oplus \Lambda_m \\ Q \wedge P = 0}}' \bar{c}_k(Q,P) \mathcal{P}_{abcd}^{(*)} e^{-2\pi R \mathcal{M}(Q,P) + 2\pi i (a_1 \cdot Q + a_2 \cdot P)} \\
& + \sum_{\substack{M_1 \neq 0, M_2 \in \mathbb{Z} \\ P \in \Lambda_m}} F_{abcd\, M_1}^{(\text{TN})} e^{-2\pi R^2 |M_1| + 2\pi i M_1},
\end{aligned} \qquad (2.21)$$

where we used the same schematic notation $\mathcal{P}_{abcd}^{(*)}$ for the tensor appearing in front of the exponential including subleading terms. The first line in (2.21) reproduces the four-dimensional couplings (2.18), while the second line corresponds to $\mathcal{O}(e^{-R})$ corrections from four-dimensional 1/2-BPS states whose wordline winds around the circle. These contributions are weighted by the BPS index $\bar{c}_k(Q,P) = \overline{\Omega}_4(Q,P)$ given in (2.16),

$$\bar{c}_k(Q,P) = \sum_{\substack{d \geq 1 \\ (Q,P)/d \in \Lambda_e \oplus \Lambda_m}} c_k\left(-\frac{\gcd(NQ^2, P^2, Q \cdot P)}{2d^2}\right) + \sum_{\substack{d \geq 1 \\ (Q,P)/d \in \Lambda_m \oplus N\Lambda_e}} c_k\left(-\frac{\gcd(NQ^2, P^2, Q \cdot P)}{2Nd^2}\right). \qquad (2.22)$$

The last line in (2.21) corresponds to $\mathcal{O}(e^{-R^2})$ corrections from Taub-NUT instantons.

Our interest in this work is on the six-derivative coupling $\nabla^2(\nabla\phi)^4$ in the low energy effective action in $D = 3$ (see (3.6) for the precise tensorial structure). The coefficient $G_{ab,cd}^{(2k,8)}(\phi)$ multiplying this coupling is valued in a vector bundle over the Grassmannian $G_{2k,8}$ associated to the representation ⊞ of $O(2k)$. As announced in [13], and proven in §3, supersymmetric Ward identities require that $G_{ab,cd}^{(2k,8)}(\phi)$ satisfies the following differential constraints

$$\mathcal{D}_{[a_1}{}^{\hat{a}}G_{a_2|b|,a_3]c}^{(2k,8)} = 0, \tag{2.23}$$

$$\mathcal{D}_{[a_1}{}^{[\hat{a}_1}\mathcal{D}_{a_2}{}^{\hat{a}_2]}G_{a_3]b,cd}^{(2k,8)} = 0, \tag{2.24}$$

$$\mathcal{D}_{[a_1}{}^{[\hat{a}_1}\mathcal{D}_{a_2}{}^{\hat{a}_2}\mathcal{D}_{a_3]}{}^{\hat{a}_3]}G_{cd,ef}^{(2k,8)} = 0, \tag{2.25}$$

$$\mathcal{D}_{(e}{}^{\hat{a}}\mathcal{D}_{f)\hat{a}}G_{ab,cd}^{(2k,8)} = -\tfrac{5}{2}\delta_{ef}G_{ab,cd}^{(2k,8)} - \left(\delta_{e)(a}G_{b)(f,cd}^{(2k,8)} + \delta_{e)(c}G_{d)(f,ab}^{(2k,8)}\right)$$
$$+ \tfrac{3}{2}\delta_{\langle ab,}G_{cd\rangle,ef}^{(2k,8)} - \tfrac{3\pi}{2}F_{|e)\langle ab,}^{(2k,8)}{}^{g}F_{cd\rangle(f|g}^{(2k,8)}. \tag{2.26}$$

Here, for two symmetric tensors $A_{ab}, B_{cd}$, we denote the projection of their product on the representation ⊞ by

$$A_{\langle ab,}B_{cd\rangle} = \tfrac{1}{3}\left(A_{ab}B_{cd} + A_{cd}B_{ab} - 2A_{|a)(c}B_{d)(b|}\right). \tag{2.27}$$

The inhomogeneous term in the last equation (2.26), proportional to the square of the $(\nabla\phi)^4$ coupling, originates from higher-derivative corrections to the supersymmetry variations. [4] It follows from (3.16)–(3.20) that in heterotic perturbation string theory, $G_{ab,cd}^{(2k,8)}$ can only receive tree-level, one-loop and two-loop corrections, plus non-perturbative corrections of order $e^{-1/g_3^2}$. We calculate the one-loop and two-loop contributions in Appendix B using earlier results in the literature [42–47]. After rescaling to Einstein frame, we find that the perturbative corrections take the form [5]

$$g_3^6 G_{ab,cd}^{(2k,8)} = -\frac{3}{4\pi g_3^2}\delta_{\langle ab,}\delta_{cd\rangle} - \frac{1}{4}\delta_{\langle ab,}G_{cd\rangle}^{(2k-1,7)} + g_3^2 G_{ab,cd}^{(2k-1,7)} + \mathcal{O}(e^{-1/g_3^2}), \tag{2.28}$$

where $G_{ab}^{(p,q)}$ denotes the genus-one modular integral

$$G_{ab}^{(p,q)} = \text{R.N.}\int_{\Gamma_0(N)\backslash\mathcal{H}_1}\frac{d\rho_1 d\rho_2}{\rho_2^2}\frac{\hat{E}_2\,\Gamma_{\Lambda_{p,q}}[P_{ab}]}{\Delta_k}, \tag{2.29}$$

with $P_{ab} = Q_{L,a}Q_{L,b} - \frac{\delta_{ab}}{4\pi\rho_2}$ and $\hat{E}_2 = E_2 - \frac{3}{\pi\rho_2}$ is the almost holomorphic Eisenstein series of weight 2, while $G_{ab,cd}^{(p,q)}$ the genus-two modular integral (of which (1.7) is a special case),

$$G_{ab,cd}^{(p,q)} = \text{R.N.}\int_{\Gamma_{2,0}(N)\backslash\mathcal{H}_2}\frac{d^3\Omega_1 d^3\Omega_2}{|\Omega_2|^3}\frac{\Gamma_{\Lambda_{p,q}}^{(2)}[P_{ab,cd}]}{\Phi_{k-2}}. \tag{2.30}$$

Here $P_{ab,cd}$ is the quartic polynomial

$$P_{ab,cd} = \varepsilon_{rt}\varepsilon_{su}Q_{L(a}^r Q_{Lb}^s Q_{L(c}^t Q_{Ld)}^u - \frac{3}{4\pi|\Omega_2|}\delta_{\langle ab,}Q_{Lc}^r(\Omega_2)_{rs}Q_{Ld\rangle}^s + \frac{3}{16\pi^2|\Omega_2|}\delta_{\langle ab,}\delta_{cd\rangle}$$
$$= \tfrac{3}{2}\left(\delta_{\langle rs,}\delta_{tu\rangle}Q_{La}^r Q_{Lb}^s Q_{Lc}^t Q_{Ld}^u - \frac{1}{2\pi|\Omega_2|}\delta_{\langle ab,}Q_{Lc}^r(\Omega_2)_{rs}Q_{Ld\rangle}^s + \frac{1}{8\pi^2|\Omega_2|}\delta_{\langle ab,}\delta_{cd\rangle}\right), \tag{2.31}$$

---

[4]Note that the properly normalized coupling in the Lagrangian is in fact $\frac{1}{\pi}G_{ab,cd}^{(2k,8)}(\phi)$, which accounts for the factor of $\pi$ on the r.h.s. of (2.26).

[5]The tree-level term comes from the double-trace contribution in [48]. The relative coefficients of the three contributions are determined by the differential equation required by the supersymmetry Ward identity, which also ensures that there are no contributions at higher loop order.

and for any polynomial $P$ in $Q^r_{La}$ and integer lattice $\Lambda_{p,q}$ of signature $(p,q)$, we denote

$$\Gamma^{(2)}_{\Lambda_{p,q}}[P] = |\Omega_2|^{q/2} \sum_{Q \in \Lambda_{p,q} \oplus \Lambda_{p,q}} P(Q_{La}) e^{i\pi Q^r_{La} \Omega_{rs} Q^{s\,a}_L - i\pi Q^r_{R\hat{a}} \bar{\Omega}_{rs} Q^{s\,\hat{a}}_R}, \qquad (2.32)$$

where $r, s = 1, 2$ label the choice of A-cycle on the genus-two Riemann surface. The regularization prescription used in defining (2.29) and (2.30) is detailed in §B.1.3 and §B.2.4, respectively.

Since the modular integral (2.30) itself satisfies the differential constraints (2.23)–(2.26), as shown in §3.3, it is consistent with supersymmetry to propose that the exact coefficient of the $\nabla^2(\nabla\phi)^4$ coupling be given by (1.7). In §§4 below, we shall demonstrate that the weak coupling expansion of the Ansatz (1.7) indeed reproduces the perturbative corrections (2.28), up to $\mathcal{O}(e^{-1/g_3^2})$ corrections. Unlike the $(\nabla\phi)^4$ couplings (2.20) however, the latter also affect the constant term in the Fourier expansion with respect to the axions $a^I$, as required by the quadratic source term in the differential equation (3.20). Such corrections can be ascribed to (NS5, KK, H-monopoles) instanton anti-instanton of vanishing total charge.

In the large radius limit, the $\nabla^2(\nabla\phi)^4$ coupling must reduce to the exact $\mathcal{R}^2 F^2$ and $\nabla^2 F^4$ couplings in $D = 4$. Consistently with this expectation, we shall find that the asymptotic expansion of (1.7) in the limit $R \to \infty$ takes the form

$$
\begin{aligned}
G^{(2k,8)}_{ab,cd} =\, & R^4 G^{(D=4)}_{ab,cd} + \frac{\zeta(3)}{8\pi}(k-12)R^6 \delta_{\langle\alpha\beta,}\delta_{\gamma\delta\rangle} \\
& + \sum_{\substack{(Q,P)\in\Lambda_e\oplus\Lambda_m \\ Q\wedge P=0}}^{\prime} \delta_{\langle ab,} \bar{G}^{(2k-1,7)}_{cd\rangle}(Q,P,t) e^{-2\pi R\mathcal{M}(Q,P)+2\pi i(a_1\cdot Q+a_2\cdot P)} \\
& + \sum_{\substack{(Q,P)\in\Lambda_e\oplus\Lambda_m \\ Q\wedge P\neq 0}}^{\prime} \bar{C}_{k-2}(Q,P;t) \mathcal{P}^{(*)}_{abcd} e^{-2\pi R\mathcal{M}(Q,P)+2\pi i(a_1\cdot Q+a_2\cdot P)} \\
& + \sum_{\substack{(Q_1,P_1)\in\Lambda_e\oplus\Lambda_m \\ Q_1\wedge P_1=0}}^{\prime} \sum_{\substack{(Q_2,P_2)\in\Lambda_e\oplus\Lambda_m \\ Q_2\wedge P_2=0}}^{\prime} \bar{c}_k(Q_1,P_1)\bar{c}_k(Q_2,P_2) \mathcal{P}^{(*)}_{abcd}(Q_1,P_1;Q_2,P_2;t) \\
& \qquad \times e^{-2\pi R[\mathcal{M}(Q_1,P_1)+\mathcal{M}(Q_2,P_2)]+2\pi i(a_1\cdot(Q_1+Q_2)+a_2\cdot(P_1+P_2))} \\
& + \sum_{M_1\neq 0} G^{(\mathrm{TN})}_{ab,cd\,M_1} e^{-2\pi R^2|M_1|+2\pi i M_1\psi} + G^{(I\bar{I})}_{ab,cd}.
\end{aligned}
\qquad (2.33)
$$

In the first line, $G^{(D=4)}_{ab,cd}$ predicts the exact $\mathcal{R}^2 F^2$ and $\nabla^2 F^4$ couplings in $D = 4$, which are exhibited in (5.67),(5.70) below, and involve explicit modular functions of the axio-dilaton $S$, as well as genus-two and genus-one modular integrals for the lattice $\Lambda_m$. These couplings are by construction invariant under the S-duality group $\Gamma_0(N)$ and under Fricke duality.

The second line in (2.33) are the 1/2-BPS Fourier coefficients, weighted by a genus-one modular integral $\bar{G}_{cd}(Q,P;t)$ for the lattice orthogonal to $Q,P$ given in (5.18), (5.46). This weighting is similar to that of 1/2-BPS contributions to the $\nabla^4\mathcal{R}^4$ coupling in maximal supersymmetric vacua [16], and is typical of Fourier coefficients of automorphic representations that do not belong to the maximal orbit in the wavefront set.

The third line corresponds to contributions from 1/4-BPS dyons, weighted by the moduli-dependent helicity supertrace, up to overall sign,

$$\bar{C}_{k-2}(Q,P;t) = (-1)^{Q\cdot P+1}\Omega_6(Q,P;t), \qquad (2.34)$$

whereas the fourth line corresponds to contributions from two-particle states consisting of two 1/2-BPS dyons that are discussed in detail in Appendices C and D. While the two contributions

on the third and fourth line are separately discontinuous as a function of the moduli $t$, their sum is continuous across walls of marginal stability. In Appendix C we show the non-trivial fact, especially for CHL orbifolds, that for fixed total charge $\Gamma$, the sum involves all possible splittings $\Gamma_1 + \Gamma_2$, weighted by the respective helicity supertraces (2.22). This complements and extends the consistency checks on the helicity supertrace formulae [30] to arbitrary charges. Moreover, we show in Appendix E that these contributions are consistent with the differential constraint (2.26). The 1/4-BPS Abelian Fourier coefficients of the non-perturbative coupling are the main focus of this paper, and the results are discussed in detail in section 5.3.

The first term $G^{(TN)}_{ab,cd\,M_1}$ on the last line corresponds to non-Abelian Fourier coefficients of order $e^{-R^2}$, ascribable to Taub-NUT instantons of charge $M_1$. We compute them in Appendix §G by dualizing the Fourier coefficients in the small coupling limit $g_3 \to 0$ computed in §4, rather than by evaluating them directly from the unfolding method.

Finally $G^{(I\bar{I})}$ contains contributions associated to instanton anti-instantons configurations,[6] which are not captured by the unfolding method but are required by the quadratic source term in the differential equation (2.26). This includes $\mathcal{O}(e^{-R})$ and $\mathcal{O}(e^{-R^2})$ contributions to the constant term, which are independent of the axions $a_1, a_2, \psi$, and contributions of order $\mathcal{O}(e^{-R^2})$ to the Abelian Fourier coefficients, which depend on the axions $a_1, a_2$ as $e^{2\pi i(a_1 \cdot Q + a_2 \cdot P)}$ but are independent of $\psi$. The latter can be ascribed to Taub-NUT instanton-anti-instantons, and are necessary in order to resolve the ambiguity of the sum over 1/4-BPS instantons [49], which is divergent due to the exponential growth of the measure $\bar{C}_{k-2}(Q, P; \Omega_2^\star) \sim (-1)^{Q \cdot P + 1} e^{\pi |Q \wedge P|}$. We do not fully evaluate $G^{(I\bar{I})}$ in this paper, but we identify the origin of the $\mathcal{O}(e^{-R^2})$ corrections as coming from poles of $1/\Phi_{k-2}$ which lie 'deep' in the Siegel upper-half plane $\mathcal{H}_2$ and do not intersect the fundamental domain, becoming relevant only after unfolding. While the precise contributions can in principle be determined by solving the differential equation (2.26), it would be interesting to obtain them via a rigorous version of the unfolding method which applies to meromorphic Siegel modular forms.

In §6, we discuss other pertubative expansions of the exact result (1.7), in the dual type I and type II pictures. In either case, the perturbative limit is dual to a large volume limit on the heterotic side, where either the full 7-torus (in the type I case) of a 4-torus (in the type II case) decompactifies. We find that the corresponding weak coupling expansion is consistent with known perturbative contributions, with non-perturbative effects associated to D-branes, NS5-branes and KK-monopoles wrapped on supersymmetric cycles of the internal space, $T^7$ in the type I case, or $K3 \times T^3$ on the type II case.

## 3  Supersymmetric Ward identities

In this section, we first establish the supersymmetric Ward identities (3.16)–(3.20) for the $\nabla^2(\nabla\phi)^4$ couplings in $D = 3$, from linearized superspace considerations. We then discuss the analogue six-derivative couplings in $D = 4$, of the form $\nabla^2 F^4$ and $\mathcal{R}^2 F^2$, and establish the corresponding Ward identities. Finally, we show that the genus-two modular integral (2.30) obeys these identities.

### 3.1  $\nabla^2(\nabla\phi)^4$ type invariants in three dimensions

This analysis is a direct generalization of the one provided in [22, §3]. We shall define the linearised superfield $W_{\hat{a}a}$ of half-maximal supergravity in three dimensions that satisfies to the

---

[6] It is worth noting that despite the fact that instanton anti-instanton configurations break all supersymmetries, they can still contribute to protected couplings, since their fermionic collective coordinates are only approximate zero-modes, lifted by interactions [50], see [51] for a cogent discussion of the instanton gas approximation.

constraints [52–54]

$$D^i_\alpha W_{\hat{a}a} = (\Gamma_{\hat{a}})^{ij}\chi_{\alpha \hat{j}a}\,, \qquad D^i_\alpha \chi_{\beta \hat{j}a} = -\mathrm{i}(\sigma^\mu)_{\alpha\beta}(\Gamma^{\hat{a}})_{\hat{j}}{}^i \partial_\mu W_{\hat{a}a}\,, \qquad (3.1)$$

with $\hat{a} = 1$ to 8 for the vector of $O(8)$, $i = 1$ to 8 for the positive chirality Weyl spinor of $Spin(8)$ and $\hat{\imath} = 1$ to 8 for the negative chirality Weyl spinor. The 1/4-BPS linearised invariants are defined using harmonics of $SO(8)/(U(2) \times SO(4))$ parametrizing a $Spin(8)$ group element $u^{r_1}{}_i, u^{r_2 r_3}{}_i, u_{r_1 i}$ in the Weyl spinor representation of positive chirality [55],

$$2u_{r_1(i}u^{r_1}{}_{j)} + \varepsilon_{r_2 s_2}\varepsilon_{r_3 s_3}u^{r_2 r_3}{}_i u^{s_2 s_3}{}_j = \delta_{ij}\,, \quad \delta^{ij}u_{r_1 i}u^{s_1}{}_j = \delta^{s_1}_{r_1}\,, \quad \delta^{ij}u_{r_1 i}u_{s_1 j} = 0\,, \qquad (3.2)$$

$$\delta^{ij}u_{r_1 i}u^{r_2 r_3}{}_j = 0\,, \quad \delta^{ij}u^{r_2 r_3}{}_i u^{s_2 s_3}{}_j = \varepsilon^{r_2 s_2}\varepsilon^{r_3 s_3}\,, \quad \delta^{ij}u^{r_1}{}_i u^{r_2 r_3}{}_j = 0\,, \quad \delta^{ij}u^{r_1}{}_i u^{s_1}{}_j = 0\,,$$

where the $r_A$ indices for $A = 1, 2, 3$ are associated to the three $SU(2)$ subgroups of $SU(2)_1 \times Spin(4) = SU(2)_1 \times SU(2)_2 \times SU(2)_3$. The harmonic variables parametrize similarly a $Spin(8)$ group element $u^{r_3 \hat{a}}, u^{r_1 r_2 \hat{a}}, u_{r_3}{}^{\hat{a}}$ in the vector representation and a group element $u^{r_2}{}_{\hat{\imath}}, u^{r_3 r_1}{}_{\hat{\imath}}, u_{r_2 \hat{\imath}}$ in the Weyl spinor representation of opposite chirality. They satisfy the same relations as (3.2) upon permutation of the three $SU(2)_A$.

The superfield $W^{r_3}_a \equiv u^{r_3 \hat{a}}W_{\hat{a}a}$ then satisfies the G-analyticity condition

$$u^{r_1}{}_i D^i_\alpha u^{r_3 \hat{a}}W_{\hat{a}a} \equiv D^{r_1}_\alpha W^{r_3}_a = 0\,. \qquad (3.3)$$

One can obtain a linearised invariant from the action of the twelve derivatives $D_{\alpha r_1} \equiv u_{r_1 i}D^i_\alpha$ and $D^{r_2 r_3}_\alpha \equiv u^{r_2 r_3}{}_i D^i_\alpha$ on any homogeneous function of the $W^{r_3}_a$'s. The integral vanishes unless the integrand includes at least the factor $W^1_{[a}W^2_{b]}W^1_{[c}W^2_{d]}$ such that the non-trivial integrands are defined as the homogeneous polynomials of degree $4+2n+m$ in $W^{r_3}_a$ in the representation of $SU(2)$ isospin $m/2$ and in the $SL(p, \mathbb{R}) \supset SO(p)$ representation of Young tableau $[n+2, m]$ ($n+2$ rows of two lines and $m$ of one line) that branches under $SO(p)$ with respect to all possible traces. After integration, the resulting expression is in the same representation of $SO(p)$ and in the irreducible representation of highest weight $m\Lambda_1 + n\Lambda_2$ of $SO(8)$, *i.e.* the traceless component associated to the Young tableau $[n, m]$, with $\Lambda_1, \Lambda_2$ denoting two fundamental weights.

It follows that the non-linear invariant only depends on the scalar fields through the tensor function $F_{ab,cd}$ and its covariant derivatives $\mathcal{D}^n F_{ab,cd}$ and covariant densities $\mathcal{L}_{[n,m]}$ in the corresponding irreducible representation of highest weight $m\Lambda_1 + n\Lambda_2$ of $SO(8)$ that only depend on the scalar fields through the covariant fields

$$P_{\mu a\hat{b}} = \partial_\mu \phi^\mu P_{\mu a\hat{b}}\,, \quad \chi_{\alpha \hat{\imath}a}\,, \quad \mathcal{D}_\mu \chi_{\alpha \hat{\imath}a} = \nabla_\mu \chi_{\alpha \hat{\imath}a} + \partial_\mu \phi^\mu \Big(\omega_{\mu a}{}^b \chi_{\alpha \hat{\imath}a} + \frac{1}{4}\omega_{\mu \hat{a}\hat{b}}(\Gamma^{\hat{a}\hat{b}})_{\hat{\imath}}{}^{\hat{\jmath}}\chi_{\alpha \hat{\jmath}a}\Big)\,, \quad (3.4)$$

and the dreibeins and the gravitini fields, and where

$$P_{a\hat{b}} \equiv \mathrm{d}p_{R\hat{b}}{}^I \eta_{IJ}p_{La}{}^J\,, \qquad \omega_{ab} \equiv -\mathrm{d}p_{La}{}^I \eta_{IJ}p_{Lb}{}^J\,, \qquad \omega_{\hat{a}\hat{b}} \equiv \mathrm{d}p_{R\hat{a}}{}^I \eta_{IJ}p_{R\hat{b}}{}^J\,, \qquad (3.5)$$

are defined from the Maurer–Cartan form of $SO(p,8)/(SO(p) \times SO(8))$. Using the known structure of the $t_8 \mathrm{tr}\nabla_\mu F \nabla^\mu F \mathrm{tr} FF$ invariant in ten dimensions [48],[7] one computes that the first covariant density $\mathcal{L}_{[0,0]}$ bosonic component is

$$
\begin{aligned}
\mathcal{L}^{ab,cd} = \frac{\sqrt{-g}}{8\pi}\Big( & 2P^{[a}_{(\mu\ \hat{a}}\nabla_\sigma P^{b]\hat{a}}_{\nu)}P^{\mu[c}_{\hat{b}}\nabla^\sigma P^{\nu|d]\hat{b}} + 2P^{[a}_{\mu(\hat{a}}\nabla_\sigma P^{\mu|b]}_{\hat{b})}P^{\nu[c|\hat{a}}\nabla^\sigma P^{d]\hat{b}}_\nu \\
& - P^{[a}_{\mu\ \hat{a}}\nabla_\sigma P^{\mu|b]\hat{a}}P^{[c}_{\nu\ \hat{b}}\nabla^\sigma P^{\nu|d]\hat{b}} - 4P^{[a|\hat{a}}_{[\mu}\nabla_\sigma P^{b]}_{\nu]}P^{\mu[c}_{[\hat{a}}\nabla^\sigma P^{\nu|d]}_{\hat{b}]} + \cdots \Big)\,.
\end{aligned}
\qquad (3.6)
$$

---

[7]with $t_8 F^4 = F_{\mu\nu}F^{\nu\sigma}F_{\sigma\rho}F^{\rho\mu} - 1/4(F_{\mu\nu}F^{\mu\nu})^2$.

The factor of $\pi$ is introduced by convenience for the definition (2.30) to hold. Investigating the possible tensors one can write in this mass dimension, one concludes that the tensor densities $\mathcal{L}_{[n,m]}$ are only non-zero for $0 \le n \le 2$ and $0 \le m \le 4$ and the density $\mathcal{L}_{[2,4]} \sim \chi^{12}$ with open $SO(p)$ indices in the symmetrization ⊞⊞⊞. The invariant admits therefore the decomposition

$$
\begin{aligned}
\mathcal{L} \;=\;& F_{ab,cd}\mathcal{L}^{ab,cd} + \mathcal{D}_e{}^{\hat{a}} F_{ab,cd}\mathcal{L}_{\hat{a}}^{ab,cd,e} + \mathcal{D}_{(e}{}^{(\hat{a}}\mathcal{D}_{f)}{}^{\hat{b})} F_{ab,cd}\mathcal{L}_{\hat{a},\hat{b}}^{ab,cd,e,f} + \mathcal{D}_{[e}{}^{[\hat{a}}\mathcal{D}_{f]}{}^{\hat{b}]} F_{ab,cd}\mathcal{L}_{\hat{a}\hat{b}}^{ab,cd,ef} \\
&+\cdots + \mathcal{D}_{(b_1}{}^{(\hat{b}_1}\cdots\mathcal{D}_{b_4)}{}^{\hat{b}_4)}\mathcal{D}_{a_1}{}^{\hat{a}_1}\cdots\mathcal{D}_{a_4}{}^{\hat{a}_4} F_{a_5a_6,a_7a_8}\mathcal{L}_{\hat{a}_1\hat{a}_2,\hat{a}_3\hat{a}_4,\hat{b}_1,\hat{b}_2,\hat{b}_3,\hat{b}_4}^{a_1a_2,a_3a_4,a_5a_6,a_7a_8,b_1,b_2,b_3,b_4},
\end{aligned}
\tag{3.7}
$$

where the $\mathcal{L}_{\hat{a}_1\hat{a}_2,\dots,\hat{a}_{2n-1}\hat{a}_{2n},\hat{b}_1,\dots,\hat{b}_m}^{a_1a_2,\dots,a_{2n+3}a_{2n+4},b_1,\dots,b_m}$ are in the irreducible representation of highest weight $m\breve{\alpha}_1 + n\breve{\alpha}_2$ of $SO(8)$ and admit the symmetry of the Young tableau $[n+2,m]$ with respect to the permutation of the $SO(p)$ indices. In particular, $F_{ab,cd}$ transforms according to ⊞, realized by first symmetrizing along the columns and then antisymmetrizing along the rows $[ab],[cd]$.

Checking the supersymmetry invariance (modulo a total derivative) of $\mathcal{L}$ in this basis, one finds that there is no term to cancel the supersymmetry variation

$$
\delta F_{ab,cd} = \big(\overline{\epsilon}_i(\Gamma^{\hat{f}})^{ij}\chi_{\hat{j}e}\big)\mathcal{D}_e{}^{\hat{f}} F_{ab,cd}
\tag{3.8}
$$

of the tensor $F_{ab,cd}$ and of its derivative when three open $SO(p)$ indices are antisymmetrized, hence the tensor $F_{ab,cd}$ must satisfy the constraints

$$
\mathcal{D}_{[a_1}{}^{\hat{a}} F_{a_2a_3],bc} = 0 \;, \quad \mathcal{D}_{[a_1}{}^{[\hat{a}_1}\mathcal{D}_{a_2}{}^{\hat{a}_2]}F_{a_3]b,cd} = 0 \;, \quad \mathcal{D}_{[a_1}{}^{[\hat{a}_1}\mathcal{D}_{a_2}{}^{\hat{a}_2}\mathcal{D}_{a_3]}{}^{\hat{a}_3]}F_{cd,ef} = 0 \;.
\tag{3.9}
$$

Similarly, because the $\mathcal{L}_{[n,m]}$ are traceless in the $SO(8)$ indices, the $SO(8)$ singlet component of $\delta(\mathcal{D}F)\mathcal{L}_{[0,1]}$ can only be cancelled by terms coming from $F\delta\mathcal{L}_{[0,0]}$, i.e.

$$
F_{ab,cd}\delta\mathcal{L}^{ab,cd} + \frac{1}{8}\mathcal{D}_e{}^{\hat{a}}\mathcal{D}_{f\hat{a}}F_{ab,cd}(\overline{\epsilon}\,\Gamma^{\hat{c}}\chi^e)\mathcal{L}_{\hat{c}}^{ab,cd,f} \sim 0
\tag{3.10}
$$

modulo terms arising from the supercovariantization, so that the covariant components must satisfy

$$
\delta\mathcal{L}^{ab,cd} + \frac{b_1}{4}(\overline{\epsilon}\,\Gamma^{\hat{c}}\chi_e)\mathcal{L}_{\hat{c}}^{ab,cd,e} + \frac{b_2}{2}\Big((\overline{\epsilon}\,\Gamma^{\hat{c}}\chi^{[a})\mathcal{L}_{\hat{c}}^{b]e,cd,}{}_e + (\overline{\epsilon}\,\Gamma^{\hat{c}}\chi^{[c})\mathcal{L}_{\hat{c}}^{d]e,ab,}{}_e\Big) = \nabla_\mu(\dots) \;.
\tag{3.11}
$$

Therefore, the tensor $F_{ab,cd}$ must obey an equation of the form

$$
\begin{aligned}
\mathcal{D}_e{}^{\hat{a}}\mathcal{D}_{f\hat{a}}F_{ab,cd} =\;& b_1\big(-\delta_{ef}F_{ab,cd} + \delta_{e[a}F_{b]f,cd} + \delta_{e[c}F_{d]f,ab}\big) \\
&- 3b_2\big(\delta_{f[a}F_{b]e,cd} + \delta_{f[c}F_{d]e,ab}\big) - 4b_2\delta_{c][a}F_{b](e,f)[d} \;,
\end{aligned}
\tag{3.12}
$$

for some numerical constants $b_1, b_2$ which are fixed by consistency. In particular the integrability condition on the component antisymmetric in $e$ and $f$ implies $b_1 = 4 - 3b_2$.

Before determining the constants $b_i$, it is convenient to generalize $F_{ab,cd}$ to a tensor $F_{ab,cd}^{(p,q)}$ on a general Grassmanian $G_{p,q}$, which would arise by considering a superfield in $D = 10 - q$ dimensions with $4 \le q \le 6$, with harmonics parametrizing $SO(q)/(U(2)\times SO(q-4))$ [56]. The same argument leads again to the conclusion that $F_{ab,cd}^{(p,q)}$ satisfies to (3.12) with $b_1 = \frac{q}{2} - 3b_2$. Equivalently, these constraints follow from the general Ansatz preserving the symmetry ⊞ of the indices $ab,cd$ and the two first equations in (3.9). An additional integrability condition comes from the equation

$$
\begin{aligned}
\mathcal{D}_{[a_1}{}^{\hat{a}}\mathcal{D}_e{}^{\hat{b}}\mathcal{D}_{|a_2|\hat{b}}F_{a_3]b,cd}^{(p,q)} =\;& [\mathcal{D}_{[a_1}{}^{\hat{a}},\mathcal{D}_e{}^{\hat{b}}]\mathcal{D}_{|a_2|\hat{b}}F_{a_3]b,cd}^{(p,q)} + \frac{1}{2}\mathcal{D}_{e\hat{b}}[\mathcal{D}_{[a_1}{}^{\hat{a}},\mathcal{D}_{a_2}{}^{\hat{b}}]F_{a_3]b,cd}^{(p,q)} \\
=\;& \mathcal{D}_{[a_1}{}^{\hat{a}}\Big(\frac{6b_2-q}{4}\delta_{e|a_2}F_{a_3]b,cd}^{(p,q)} + \frac{3b_2}{2}\delta_{b|a_2}F_{a_3]e,cd}^{(p,q)} + 2b_3\delta_{c||a_2}F_{a_3]b,e[d}^{(p,q)} + b_3\delta_{c||a_2}F_{a_3][d,be}^{(p,q)}\Big) \\
=\;& \mathcal{D}_{[a_1}{}^{\hat{a}}\Big(\frac{3-q}{4}\delta_{e|a_2}F_{a_3]b,cd}^{(p,q)} + \frac{1}{4}\delta_{b|a_2}F_{a_3]e,cd}^{(p,q)} + \frac{1}{2}\delta_{c||a_2}F_{a_3]b,e[d}^{(p,q)}\Big) \\
&+ \frac{1}{4}\mathcal{D}_e{}^{\hat{a}}\Big(\delta_{b[a_1}F_{a_2a_3],cd}^{(p,q)} + \delta_{c][a_1}F_{a_2a_3],b[d}^{(p,q)}\Big) \;,
\end{aligned}
\tag{3.13}
$$

which is indeed consistent, if and only if $b_2 = \frac{1}{2}$ and so $b_1 = \frac{q-3}{2}$ so that (3.12) reduces to

$$\mathcal{D}_{(e}{}^{\hat{a}}\mathcal{D}_{f)\hat{a}}F^{(p,q)}_{ab,cd} = \tfrac{3-q}{2}\delta_{ef}F^{(p,q)}_{ab,cd} + \tfrac{q-6}{2}\big(\delta_{e)[a}F^{(p,q)}_{b](f,cd} + \delta_{e)[c}F^{(p,q)}_{d](f,ab}\big) - 2\delta_{c][a}F^{(p,q)}_{b](e,f)[d} . \quad (3.14)$$

Alternatively, one can represent a tensor with the symmetry $\boxplus$ with two pairs of indices that are manifestly symmetric, *i.e.* $G_{ab,cd} = G_{ba,cd} = G_{ab,dc} = G_{cd,ab}$ such that $G_{(ab,c)d} = 0$, such that

$$F_{ab,cd} = G_{c][a,b][d} , \qquad G_{ab,cd} = -\tfrac{4}{3}F_{a)(c,d)(b} . \quad (3.15)$$

The tensor $G_{ab,cd}$ satisfies the constraints

$$\mathcal{D}_{[a_1}{}^{\hat{a}}G^{(p,q)}_{a_2|b|,a_3]c} = 0 , \quad \mathcal{D}_{[a_1}{}^{[\hat{a}_1}\mathcal{D}_{a_2}{}^{\hat{a}_2]}G^{(p,q)}_{a_3]b,cd} = 0 , \quad \mathcal{D}_{[a_1}{}^{[\hat{a}_1}\mathcal{D}_{a_2}{}^{\hat{a}_2}\mathcal{D}_{a_3]}{}^{\hat{a}_3]}G^{(p,q)}_{cd,ef} = 0 , \quad (3.16)$$

and

$$\mathcal{D}_{(e}{}^{\hat{a}}\mathcal{D}_{f)\hat{a}}G^{(p,q)}_{ab,cd} = \tfrac{3-q}{2}\delta_{ef}G^{(p,q)}_{ab,cd} + \tfrac{6-q}{2}\big(\delta_{e)(a}G^{(p,q)}_{b)(f,cd} + \delta_{e)(c}G^{(p,q)}_{d)(f,ab}\big) + \tfrac{3}{2}\delta_{\langle ab,}G^{(p,q)}_{cd\rangle,ef} . \quad (3.17)$$

The discussion so far only applies to a supersymmetry invariant modulo the classical equations of motion, whereas one must take into account the first correction in $(\nabla\phi)^4$. The direct computation of this correction via supersymmetry invariance at the next order is extremely difficult, however, one can determine its form from general arguments. The modification of the supersymmetry Ward identities implies that the corrections to the differential equations must be an additional source term quadratic in the completely symmetric tensor $F^{(p,q)}_{abcd}$ defining the $(\nabla\phi)^4$ coupling. This correction should preserve the wave-front set associated to the original homogeneous solution, so it is expected that (3.16) is not modified, while the second order equation (3.17) admits a source term quadratic in $F^{(p,q)}_{abcd}$ and consistent with (3.16). Inspection of the various possible tensor structures shows that there is indeed no possible correction to (3.16), because $F^{(p,q)}_{abcd}$ satisfies itself

$$\mathcal{D}_{[a}{}^{\hat{a}}F^{(p,q)}_{b]cde} = 0 , \qquad \mathcal{D}_{[a}{}^{[\hat{a}}\mathcal{D}_{b]}{}^{\hat{b}]}F^{(p,q)}_{cdef} = 0 . \quad (3.18)$$

Equation (3.17) admits the symmetry associated to the Young tableaux $\boxplus\!\square$ and $\boxplus^{\square}$, however it is easy to check that the latter is trivially satisfied

$$\tfrac{1}{2}\mathcal{D}_{[a_1|}{}^{\hat{a}}\mathcal{D}_{b\hat{a}}F^{(p,q)}_{|a_2a_3],cd} = -\tfrac{q}{4}\delta_{b[a_1}F^{(p,q)}_{a_2a_3],cd} - \tfrac{q}{4}\delta_{c][a_1}F^{(p,q)}_{a_2a_3],b[d} , \quad (3.19)$$

and therefore cannot be corrected by a source term. The only source term quadratic in $F^{(p,q)}_{abcd}$ with the symmetry structure $\boxplus\!\square$ that also satisfies to the constraint (3.16) is $F^{(p,q)}_{|e)\langle ab,}{}^{g}F^{(p,q)}_{cd)(f|g}$. It is indeed straighforward to check that the corresponding combination sourcing (3.17), namely $F^{(p,q)}_{c]e[a}{}^{g}F^{(p,q)}_{b]fg[d}$, satisfies (3.9) using (3.18), whereas any other combination with the symmetry structure $\boxplus\!\square$ involving the Kronecker symbol would not.

We conclude that the correct supersymmetry constraint for $G^{(p,q)}_{ab,cd}$ reads

$$\mathcal{D}_{(e}{}^{\hat{a}}\mathcal{D}_{f)\hat{a}}G^{(p,q)}_{ab,cd} = \tfrac{3-q}{2}\delta_{ef}G^{(p,q)}_{ab,cd} + \tfrac{6-q}{2}\big(\delta_{e)(a}G^{(p,q)}_{b)(f,cd} + \delta_{e)(c}G^{(p,q)}_{d)(f,ab}\big) + \tfrac{3}{2}\delta_{\langle ab,}G^{(p,q)}_{cd\rangle,ef}$$
$$- \tfrac{3\varpi}{2}F^{(p,q)}_{|e)\langle ab,}{}^{g}F^{(p,q)}_{cd)(f|g} , \quad (3.20)$$

where $\varpi$ is an undetermined numerical coefficient at this stage. In §3.3 we shall show that the genus-two modular integral (2.30) satisfies this equation with $\varpi = \pi$.

Let us note that this discussion only applies to the Wilsonian effective action. As we shall see in section B.2.4, the differential Ward identity satisfied by the renormalized coupling $\hat{G}_{ab,cd}$

appearing in the 1PI effective action is expected to be corrected in four dimensions ($q = 6$) by constant terms and by terms linear in $\hat{F}_{abcd}$.

Because of the quadratic source term in (3.20), the tensor $G_{ab,cd}$ does not belong strictly speaking to an automorphic representation of $SO(p,q)$. One can nonetheless define a generalization of the notion of automorphic representation attached to this tensor. The linearised analysis exhibits that the homogeneous differential equation is attached to the $SO(p,q)$ representation associated to the nilpotent orbit of partition $[3^2, 1^{p+q-6}]$ such that the nilpotent elements $Z_{a\hat{b}} \in \mathfrak{so}(p+q)(\mathbb{C}) \ominus (\mathfrak{so}(p)(\mathbb{C}) \oplus \mathfrak{so}(q)(\mathbb{C}))$ satisfy the constraint (cf. (3.9), (3.12))

$$Z_{[a}{}^{[\hat{a}} Z_b{}^{\hat{b}} Z_{c]}{}^{\hat{c}]} = 0 , \qquad Z_{a\hat{c}} Z_b{}^{\hat{c}} = 0 . \tag{3.21}$$

For a representative of the nilpotent orbit in the unipotent associated to the maximal parabolic $GL(k) \times SO(p-k, q-k) \ltimes \mathbb{R}^{2(p+q-2k) + \frac{k(k-1)}{2}}$ this gives the constraints [8]

$$Q_{[i}{}^{[m} Q_j{}^n Q_{k]}{}^{p]} = 0 , \qquad Q_{[i}{}^m Q_j{}^n K_{kl]} = 0, \tag{3.22}$$

which admits a subspace of solutions of dimension $2(p + q - k - 2)$ for $Q_i{}^m \in SL(k) \times SO(p-k, q-k)/(SO(2) \times SL(k-2) \ltimes \mathbb{R}^{2(k-2)} \times SO(p-k-2, q-k)) \in \mathbb{R}^{2(p+q-2k)}$ and a subspace of dimension $2k-3$ for $K_{ij} \in \mathbb{R}^{\frac{k(k-1)}{2}}$, and therefore a Kostant–Kirillov dimension $2(p+q-4)+1$ that is exactly saturated by the Fourier coefficients in the maximal parabolic decomposition with $k = 2$.

The tensor $F_{abcd}$ is instead in an automorphic representation associated to the nilpotent orbit of partition $[3, 1^{p+q-3}]$ such that the nilpotent elements $Z_{a\hat{b}} \in \mathfrak{so}(p+q)(\mathbb{C}) \ominus (\mathfrak{so}(p)(\mathbb{C}) \oplus \mathfrak{so}(q)(\mathbb{C}))$ satisfy the constraint

$$Z_{[a}{}^{[\hat{a}} Z_{b]}{}^{\hat{b}]} = 0 , \qquad Z_{a\hat{c}} Z_b{}^{\hat{c}} = 0 . \tag{3.23}$$

For a representative of the nilpotent orbit in the unipotent associated to the maximal parabolic $GL(k) \times SO(p-k, q-k) \ltimes \mathbb{R}^{2(p+q-2k) + \frac{k(k-1)}{2}}$ this gives the constraints

$$Q_{[i}{}^{[m} Q_{j]}{}^{n]} = 0 , \qquad Q_{[i}{}^m K_{jk]} = 0 , \qquad K_{[ij} K_{kl]} = 0 , \tag{3.24}$$

which admits a subspace of solutions of dimension $p + q - k - 1$ for $Q_i{}^m \in SL(k) \times SO(p-k, q-k)/(SL(k-1) \ltimes \mathbb{R}^{k-1} \times SO(p-k-1, q-k)) \in \mathbb{R}^{2(p+q-2k)}$ and a subspace of dimension $k-1$ for $K_{ij} \in \mathbb{R}^{\frac{k(k-1)}{2}}$, and therefore a Kostant–Kirillov dimension $p+q-2$ that is exactly saturated by the Fourier coefficients in the maximal parabolic decomposition with $k = 1$. One easily checks that the sum of two generic elements $(Q_i^m, K_{ij})$ solving (3.24) always solve (3.22), so that the quadratic source in $F_{abcd}$ sources the Fourier coefficients of the tensor $G_{ab,cd}$ consistently with the automorphic representation associated to the nilpotent orbit of partition $[3^2, 1^{p+q-6}]$.

It is important to note that the 1/4-BPS black hole solutions (single-centered and multi-centered) are solutions of the Euclidean three-dimensional non-linear sigma model over $O(2k, 8)/(O(2k) \times O(8))$ which are themselves associated to a real nilpotent orbit of $O(2k, 8)$ of partition $[3^2, 1^{2+2k}]$ [57, 58]. This is consistent with the property that the Fourier coefficients in the maximal parabolic decomposition $GL(2) \times O(2k-2, 6) \ltimes \mathbb{R}^{2(4+2k)+1}$ saturate the Kostant–Kirillov dimension and are proportional to the helicity supertrace associated with these black holes.

---

[8]The unipotent being non-Abelian for $k \geq 2$, one cannot generally define the Fourier coefficients for $(Q_i^m, K_{ij})$, but one must consider separately the Abelian Fourier coefficient with $K_{ij} = 0$, from the non-Abelian Fourier coefficients with $K_{ij}$ and a subset of the charges $Q_i^m$ defining a polarization.

## 3.2 $\mathcal{R}^2 F^2$ type invariants in four dimensions

In four dimensions, there are two distinct classes of six-derivative supersymmetric invariants. In the linearised approximation, they are defined as harmonic superspace integrals of G-analytic integrands annihilated by a quarter of the fermionic derivatives, and can be promoted to non-linear harmonic superspace integrals [59]. The first class of invariants is the one defined in the preceding section for $q = 6$. It includes a $G^{(2k-2,6)}_{ab,cd} \nabla(F^a \bar{F}^b) \nabla(F^c \bar{F}^d)$ coupling with a tensor $G^{(2k-2,6)}_{ab,cd}$ satisfying to (3.16) and (3.20). The second class of invariants is defined as a chiral harmonic superspace integral at the linearised level, as we now explain.

In four dimensional supergravity with half-maximal supersymmetry, the linearised Maxwell superfield $W_{\hat{a}a} \sim W_{ija}$ satisfies the constraints

$$D_{\alpha k} W_{ija} = \varepsilon_{ijkl} \lambda^l_{\alpha a} , \qquad \bar{D}^k_{\dot{\alpha}} W_{ija} = 2\delta^k_{[i} \bar{\lambda}_{\dot{\alpha}j]a} , \qquad D_{\alpha i} \lambda^j_{\beta a} = \delta^j_i F_{\alpha\beta a} , \qquad (3.25)$$

whereas the chiral scalar superfield satisfies

$$D_{\alpha i} S = \chi_{\alpha i} , \qquad \bar{D}^i_{\dot{\alpha}} S = 0 , \qquad D_{\alpha i} \chi_{\beta i} = F_{\alpha\beta ij} , \qquad (3.26)$$

with $i = 1$ to $4$ of $SU(4)$ and $\alpha, \dot{\alpha}$ the $SL(2,\mathbb{C})$ indices. The chiral 1/4-BPS linearised invariants are defined using harmonics of $SU(4)/S(U(2) \times U(2))$ parametrizing a $SU(4)$ group element $u_r{}^i, u_{\hat{r}}{}^i$ with $r$ and $\hat{r}$ the indices of the two respective $SU(2)$ subgroups. The superfield $W_{34a} \equiv u_3{}^i u_4{}^j W_{ija} = \frac{1}{2}\varepsilon^{\hat{r}\hat{s}} u_{\hat{r}}{}^i u_{\hat{s}}{}^j W_{ija}$ then satisfies the G-analyticity constraints

$$u_{\hat{r}}{}^i D_{\alpha i}(u_3{}^i u_4{}^j W_{ija}) \equiv D_{\alpha\hat{r}} W_{34a} = 0 , \qquad u_i{}^r \bar{D}^i_{\dot{\alpha}}(u_3{}^i u_4{}^j W_{ija}) \equiv \bar{D}^r_{\dot{\alpha}} W_{34a} = 0 . \qquad (3.27)$$

One can obtain a linearised invariant from the action of the eight derivatives $D_{\alpha i}$ and the four derivatives $\bar{D}^r_{\dot{\alpha}} \equiv u_i{}^r \bar{D}^i_{\dot{\alpha}}$ on any homogeneous function of the G-analytic superfields $W_{34a}$ and $S$. Using for short $u_{\hat{a}}^{34} = (\Gamma_{\hat{a}})^{ij} u_i{}^3 u_j{}^4$ and the projection $(\hat{a}_1 \ldots \hat{a}_n)'$ on the traceless symmetric component, one gets

$$
\begin{aligned}
&\int du \, u^{34}_{\hat{a}_1} \ldots u^{34}_{\hat{a}_n} [D^8][\bar{D}^4] \frac{1}{(n+2)!(m+2)!} c_{a_1 \ldots a_{n+2}} W^{a_1}_{34} \ldots W^{a_{n+2}}_{34} S^{2+m} \\
&= \frac{1}{n!m!} c_{a_1 \ldots a_n ab} W^{a_1}{}_{(\hat{a}_1} W^{a_2}{}_{\hat{a}_2} \ldots W^{a_n}{}_{\hat{a}_n)'} S^m \mathcal{L}^{(0)ab}_{+2} \\
&\quad + \frac{1}{(n-1)!m!} c_{a_1 \ldots a_n ab} W^{a_2}{}_{(\hat{a}_2} W^{a_3}{}_{\hat{a}_3} \ldots W^{a_n}{}_{\hat{a}_n)} S^m \mathcal{L}^{(0)a_1 ab}_{\hat{a}_1)'+2} \\
&\quad + \frac{1}{n!(m-1)!} c_{a_1 \ldots a_n ab} W^{a_1}{}_{(\hat{a}_1} W^{a_2}{}_{\hat{a}_2} \ldots W^{a_n}{}_{\hat{a}_n)'} S^{m-1} \mathcal{L}^{(0)ab}_{+4} + \ldots \\
&\quad + \frac{1}{(n-6)!(m-2)!} c_{a_1 \ldots a_n ab} W^{a_7}{}_{(\hat{a}_7} W^{a_6}{}_{\hat{a}_6} \ldots W^{a_n}{}_{\hat{a}_n)} S^{m-2} \mathcal{L}^{(0)a_1 \ldots a_6 ab}_{\hat{a}_1 \ldots \hat{a}_6)'+6} + \ldots \\
&\quad + \frac{1}{(n-2)!(m-6)!} c_{a_1 \ldots a_n ab} W^{a_3}{}_{(\hat{a}_3} W^{a_4}{}_{\hat{a}_4} \ldots W^{a_n}{}_{\hat{a}_n)} S^{m-6} \mathcal{L}^{(0)a_1 a_2 ab}_{\hat{a}_1 \hat{a}_2)'+14} + \partial(\ldots) , \quad (3.28)
\end{aligned}
$$

where the $\mathcal{L}^{[n+4]}_{[n]+m}$ are symmetric tensors that only depend on the scalar fields through their derivative. One works out in particular that $\mathcal{L}^{(0)ab}_{+2}$ includes a term of type $\mathcal{R}^2 F^2$ as

$$\mathcal{L}^{(0)ab}_{+2} \sim \bar{F}^a_{\dot{\alpha}\dot{\beta}} \bar{F}^{\dot{\alpha}\dot{\beta}b} C_{\alpha\beta\gamma\delta} C^{\alpha\beta\gamma\delta} + \ldots , \qquad (3.29)$$

with $C_{\alpha\beta\gamma\delta}$ the complex Weyl curvature tensor (which we denote schematically by $\mathcal{R}$), whereas the highest monomials only depend on the fermion fields as

$$
\begin{aligned}
\mathcal{L}^{(0)a_1 \ldots a_6 ab}_{\hat{a}_1 \ldots \hat{a}_6 +6} &\sim \bar{\lambda}^4 \lambda^4 \chi^4 , \qquad \mathcal{L}^{(0)a_1 \ldots a_5 ab}_{\hat{a}_1 \ldots \hat{a}_5 +8} \sim \bar{\lambda}^4 \lambda^3 \chi^5 , \qquad \mathcal{L}^{(0)a_1 \ldots a_4 ab}_{\hat{a}_1 \ldots \hat{a}_4 +10} \sim \bar{\lambda}^4 \lambda^2 \chi^6 , \\
\mathcal{L}^{(0)a_1 a_2 a_3 ab}_{\hat{a}_1 \hat{a}_2 \hat{a}_3 +12} &\sim \bar{\lambda}^4 \lambda \chi^7 , \qquad \mathcal{L}^{(0)a_1 a_2 ab}_{\hat{a}_1 \hat{a}_2 +14} \sim \bar{\lambda}^4 \chi^8 . \qquad (3.30)
\end{aligned}
$$

Note that $\mathcal{L}_{+2}^{(0)ab}$ is of $U(1)$ weight $-2$, so one can anticipate that it must be multiplied by a modular form of weight 2 at the non-linear level. At the non-linear level, derivatives of the scalar fields only appear through the pull-back of the right-invariant form $P_{a\hat{b}}$ over the Grassmanian and the covariant derivative $(S-\bar{S})^{-1}\partial_\mu S$ of the upper complex half plan field $S$. One defines in the same way the covariant derivative $\mathcal{D}_{a\hat{b}}$ on the Grassmanian and the Kähler derivative $\mathcal{D} = (S-\bar{S})\frac{\partial}{\partial S} + \frac{w}{2}$ on a weight $w$ form. According to the linearised analysis, the supersymmetry invariant is associated to a tensor $G_{ab}(\phi, S)$, holomorphic in $S$ and function of the Grassmanian coordinates $\phi$.

Due to the superconformal symmetry $PSU(2,2|4)$ of the linearised theory in four dimensions, the non-linear invariants are in bijective correspondance with the linearised invariants, themselves determined by harmonic superspace integrals. However, the linearised invariants that combine to define a general class of non-linear invariants are not necessarily defined from the same harmonic superspace. The general $\nabla^2 F^4$ type invariants defined in the preceding section are determined by vector-like harmonic superspace integrals of $SU(4)/S(U(1)\times U(2)\times U(1))$. In contrast the $\mathcal{R}^2 F^2$ type invariants described in this section involve both structures, such that the defining function $\mathcal{G}_{ab}(\phi, S)$ is of weight zero, and the terms in the Lagrangian that do not involve its Kähler derivative $\mathcal{D}$ are defined at the linearised level from $SU(4)/S(U(1)\times U(2)\times U(1))$ harmonic superspace integral of a restricted type. These invariants are constructed explicitly in [59] for a $SO(p)$ invariant function on the Grassmannian. One finds that $\mathcal{G}_{ab}(\phi, S)$ must be holomorphic in $S$, as the linearised analysis suggested. It defines a Lagrange density $\mathcal{L}$ that decomposes naturally as

$$
\begin{aligned}
\mathcal{L} = \mathcal{G}_{ab}\mathcal{L}^{ab} &+ \mathcal{D}_{(a}{}^{\hat{a}}\mathcal{G}_{bc)}\mathcal{L}^{abc}{}_{\hat{a}} + \mathcal{D}_{(a}{}^{(\hat{a}}\mathcal{D}_b{}^{\hat{b})}\mathcal{G}_{cd)}\mathcal{L}^{abcd}{}_{\hat{a}\hat{b}} + \dots \\
&+ \mathcal{D}\mathcal{G}_{ab}\mathcal{L}^{ab}_{+2} + \dots + \mathcal{D}\mathcal{D}_{(a_1}{}^{\hat{a}_1}\cdots\mathcal{D}_{a_6}{}^{\hat{a}_6}\mathcal{G}_{a_7a_8)}\mathcal{L}^{a_1\dots a_8}{}_{\hat{a}_1\dots\hat{a}_6+2} \\
&+ \mathcal{D}^2\mathcal{G}_{ab}\mathcal{L}^{ab}_{+4} + \dots + \mathcal{D}^2\mathcal{D}_{(a_1}{}^{\hat{a}_1}\cdots\mathcal{D}_{a_5}{}^{\hat{a}_5}\mathcal{G}_{a_6a_7)}\mathcal{L}^{a_1\dots a_7}{}_{\hat{a}_1\dots\hat{a}_5+4} \\
&\quad\quad\quad\quad\quad\quad\quad\quad\quad\quad\quad\vdots \\
&+ \mathcal{D}^7\mathcal{G}_{ab}\mathcal{L}^{ab}_{+14} + \mathcal{D}^7\mathcal{D}_{(a}{}^{\hat{a}}\mathcal{G}_{bc)}\mathcal{L}^{abc}{}_{\hat{a}+14} + \mathcal{D}^7\mathcal{D}_{(a}{}^{(\hat{a}}\mathcal{D}_b{}^{\hat{b})}\mathcal{G}_{cd)}\mathcal{L}^{abcd}{}_{\hat{a}\hat{b}+14},
\end{aligned}
\tag{3.31}
$$

where the $\mathcal{L}^{[n+2]}{}_{[n]+m}$ are $SL(2)\times O(2k-2,6)$ invariant polynomial functions of the covariant fields and their derivatives and the vierbeins and the gravitini fields. Because non-linear invariants induce linear invariants by truncation to lowest order in the fields (3.4), the covariant densities $\mathcal{L}^{[n+2]}{}_{[n]+m}$ reduce at lowest order to homogeneous polynomials of degree $n+2$ in the covariant fields (3.4) that coincide with the linearised polynomials $\mathcal{L}^{(0)[n+2]}{}_{[n]+m}$ for $m \geq 2$. For $m = 0$, the linearised invariants $\mathcal{L}^{(0)[n+2]}{}_{[n]}$ are the real analytic superspace integrals described in the preceding section $[n+2, m]$ for $n = 0$, and where indices are contracted with $\delta^{ab}$ to reduce the representation from the Young Tableau $[2, m]$ to $[0, m+1]$. The analysis of the invariant defined as a non-linear harmonic superspace integral indeed shows that the component $\mathcal{L}^{ab}$ is of the type

$$
\mathcal{L}^{ab} = \sqrt{-g}\, t_8\big(2\nabla(F^{(a}F^{b)})\nabla(F_c F^c) + \nabla(F_c F^{(a})\nabla(F^{b)}F^c) + \dots\big),
\tag{3.32}
$$

with $t_8 F^4 = F_{\alpha\beta}F^{\alpha\beta}\bar{F}_{\dot\alpha\dot\beta}\bar{F}^{\dot\alpha\dot\beta}$, and

$$
\mathcal{L}^{ab}_{+2} = \sqrt{-g}\,\bar{F}^a_{\dot\alpha\dot\beta}\bar{F}^{\dot\alpha\dot\beta b}C_{\alpha\beta\gamma\delta}C^{\alpha\beta\gamma\delta} + \dots.
\tag{3.33}
$$

The complete invariant is the real part of this complex invariant. So the four-photon MHV amplitude gives a contribution to the Wilsonian effective action in $\mathcal{G}_{ab}(\phi, S) + \mathcal{G}_{ab}(\phi, \bar{S})$, whereas the amplitude with two gravitons of positive helicity and two photons of negative helicity

gives a contribution in $\mathcal{D}\mathcal{G}_{ab}(\phi, S)$. Because $\mathcal{D}\mathcal{G}_{ab}(\phi, \bar{S}) = 0$, we will usually refer to a single function $G^{(0)}_{ab}(\phi, S, \bar{S}) = \mathcal{G}_{ab}(\phi, S) + \mathcal{G}_{ab}(\phi, \bar{S})$.

Similarly to [22], one can show that supersymmetry at the linearised level implies tensorial differential equations of the form

$$\mathcal{D}_d{}^{\hat{a}}\mathcal{D}_{c\hat{a}}G^{(0)}_{ab} = \tfrac{3(2-q)}{4}\delta_{d(c}G^{(0)}_{ab)} + \tfrac{3}{2}\delta_{(ab}G^{(0)}_{c)d} , \qquad \mathcal{D}_{[a}{}^{\hat{a}}G^{(0)}_{b]c} = 0 , \tag{3.34}$$

with $q = 6$, where the coefficients of the two terms on the right-hand side have been fixed by requiring that these constraints are integrable.

As in the preceding section, this linearized analysis does not take into account the lower order corrections in the effective action and the local terms coming from the explicit decomposition of the effective action into local and non-local components. The coefficient $F_{abcd}(\varphi)$ of the $F^4$ coupling and the real coefficient $\mathcal{E}(S)$ of the $\mathcal{R}^2$ coupling give rise to source terms in these differential equations, such that we get eventually

$$\begin{aligned}
\mathcal{D}_d{}^{\hat{a}}\mathcal{D}_{c\hat{a}}G_{ab}(S, \varphi) &= -3\delta_{d(c}G_{ab)}(S, \varphi) + \tfrac{3}{2}\delta_{(ab}G_{c)d}(S, \varphi) + 6\mathcal{E}(S)F_{abcd}(\varphi) , \\
\mathcal{D}\bar{\mathcal{D}}G_{ab}(S, \varphi) &= \frac{3}{4\pi}F_{abc}{}^c(\varphi) .
\end{aligned} \tag{3.35}$$

Finally, let us note that the same class of harmonic superspace integrals (3.28) produces higher derivative invariants by integrating instead

$$\int \mathrm{d}u\, u^{34}_{\hat{a}_1}\ldots u^{34}_{\hat{a}_n}u^{34}_{\hat{b}_1}\ldots u^{34}_{\hat{b}_{2p}} [D^8][\bar{D}^4]\frac{1}{n!(m+2)!(p+1)!}c_{a_1\ldots a_n}W^{a_1}_{34}\ldots W^{a_n}_{34}S^{2+m}(F_{\alpha\beta 34}F^{\alpha\beta}_{34})^{p+1} . \tag{3.36}$$

This gives rise to chiral 1/4-BPS-protected invariants of the same class, including couplings of the form

$$\mathcal{G}^{(2p+4)}_{\hat{a}_1\hat{a}_2\ldots\hat{a}_{2p}}(S, \varphi)\,C^2\,\nabla^2 S\nabla^2 S(F^{\hat{a}_1}F^{\hat{a}_2})\ldots(F^{\hat{a}_{2p-1}}F^{\hat{a}_{2p}}) . \tag{3.37}$$

Here $C$ is the Weyl tensor and $\mathcal{G}^{(2p)}(S, \varphi)$ is a rank $2p$ $SO(6)$ symmetric traceless tensor, which is a weight $2p+4$ weakly holomorphic modular form in $S$. It satisfies to a hierarchy of differential equations on the Grassmannian [60]

$$\begin{aligned}
\mathcal{D}_a{}^{\hat{a}_{2p}}\mathcal{G}^{(2p+4)}_{\hat{a}_1\ldots\hat{a}_{2p}} = \mathcal{D}_{a[\hat{a}_1}\mathcal{G}^{(2p+4)}_{\hat{a}_2]\hat{a}_3\ldots\hat{a}_{2p+1}} = \bar{\mathcal{D}}\mathcal{G}^{(2p+4)}_{\hat{a}_1\ldots\hat{a}_{2p}} = 0 , \\
\mathcal{D}_a{}^{\hat{c}}\mathcal{D}_{b\hat{c}}\mathcal{G}^{(2p+4)}_{\hat{a}_1\ldots\hat{a}_{2p}} = -2(p+2)\delta_{ab}\mathcal{G}^{(2p+4)}_{\hat{a}_1\ldots\hat{a}_{2p}} + \mathcal{D}_{a(\hat{a}_1}\mathcal{D}_{|b|\hat{a}_2}\mathcal{G}^{(2p+2)}_{\hat{a}_3\ldots\hat{a}_{2p})} .
\end{aligned} \tag{3.38}$$

On the type II side these couplings can be computed in topological string theory [61].

## 3.3 The modular integral satisfies the Ward identities

In this subsection, we shall prove that the modular integral (2.30), which we copy for convenience,

$$G^{(p,q)}_{ab,cd} = \text{R.N.}\int_{\Gamma_{2,0}(N)\backslash\mathcal{H}_2}\frac{\mathrm{d}^3\Omega_1\mathrm{d}^3\Omega_2}{|\Omega_2|^3}\frac{\Gamma^{(2)}_{\Lambda_{p,q}}[P_{ab,cd}]}{\Phi_{k-2}(\Omega)} \tag{3.39}$$

satisfies the differential equations (3.16) and (3.20), with a specific value of the coefficient $\varpi$ in the quadratic source term. Here, $\Phi_{k-2}(\Omega)$ is the meromorphic Siegel modular form defined in (A.33), and $\Gamma^{(2)}_{\Lambda_{p,q}}[P_{ab,cd}]$ is the genus-two partition function (2.32) for a level $N$ even lattice of signature $(p,q)$, with an insertion of the quartic polynomial $P_{ab,cd}$ defined in (2.31). Since $\Phi_{k-2}$ and $\Gamma^{(2)}_{\Lambda_{p,q}}[P_{ab,cd}]$ are modular forms of weight $k-2$ and $\frac{p-q}{2} + 2 = k-2$ under $\Gamma_{2,0}(N)$, the integrand is well defined on the quotient $\Gamma_{2,0}(N)\backslash\mathcal{H}_2$. The symbol R.N.

refers to a regularization procedure which is necessary to make sense of the integral when $q \geq 5$, as discussed in Appendix B.2.4.

In order to derive these results, we shall first establish differential equations for the general class of genus-two Siegel theta series $\Gamma^{(2)}_{\Lambda_{p,q}}[P]$, where the polynomial $P(Q)$ is obtained by acting on a homogeneous polynomial of bidegree $(m, n)$ in $(\varepsilon_{rs} Q_L^r Q_L^s, \varepsilon_{rs} Q_R^r Q_R)^s$ respectively, with the operator $|\Omega_2|^n e^{-\frac{\Delta_2}{8\pi}}$, where $\varepsilon_{rs}$ is the rank-two antisymmetric tensor with $\varepsilon_{12} = 1$ and $\Delta_2$ is the second order differential operator

$$\Delta_2 \equiv \sum_a \frac{\partial}{\partial Q_{La}^r} (\Omega_2^{-1})^{rs} \frac{\partial}{\partial Q_L^{sa}} + \sum_{\hat{a}} \frac{\partial}{\partial Q_{R\hat{a}}^r} (\Omega_2^{-1})^{rs} \frac{\partial}{\partial Q_R^{s\hat{a}}}. \tag{3.40}$$

Under this condition, one can show using Poisson resummation that $\Gamma^{(2)}_{\Lambda_{p,q}}[P]$ satisfies

$$\Gamma^{(2)}_{\Lambda_{p,q}}[P](-\Omega^{-1}) = \frac{(-i)^{p-q} |\Omega|^{\frac{p-q}{2} + m - n}}{|\Lambda^*_{p,q} / \Lambda_{p,q}|} \Gamma^{(2)}_{\Lambda^*_{p,q}}[P](\Omega), \tag{3.41}$$

which implies that $\Gamma^{(2)}_{\Lambda_{p,q}}[P]$ transforms as a modular form of weight $\frac{p-2}{2} + m - n$ under $\Gamma_{2,0}(N)$. For our purposes, it will be sufficient to focus on polynomials of the form, using $(Q_a \varepsilon Q_b) = \varepsilon_{rs} Q_a^r Q_b^s$,

$$
\begin{aligned}
&P_{a_1 \dots a_m, b_1 \dots b_m, \hat{c}_1 \dots \hat{c}_n, \hat{d}_1 \dots \hat{d}_n} \\
&= e^{\frac{-\Delta_2}{8\pi}} \Big[ \big( (Q_{L(a_1|} \varepsilon Q_{L|b_1|}) \dots (Q_{L|a_m|} \varepsilon Q_{Lb_m}) \big) \big( (Q_{R(\hat{c}_1|} \varepsilon Q_{R|\hat{d}_1|}) \dots (Q_{R|\hat{c}_n|} \varepsilon Q_{R\hat{d}_n}) \big) \Big],
\end{aligned} \tag{3.42}
$$

where $(b_1 \dots b_m)$ denotes all symmetric permutations of $b_1, \dots, b_m$, and similarly for hatted indices. The quadratic polynomial $P = P_{ab,cd}$ arises in the case $(m, n) = (2, 0)$ with no contraction among the left-moving indices, as written explicitly in the first line of (2.31).

As in [22, section 3], one can obtain the differential equations satisfied by (3.39) by acting with the covariant derivatives $\mathcal{D}_{a\hat{b}}$ defined by

$$\mathcal{D}_{a\hat{b}} = \frac{1}{2} (Q_{La}^r \partial_{r\hat{b}} + Q_{R,\hat{b}}^r \partial_{ra}), \tag{3.43}$$

where $\partial_r^a = \frac{\partial}{\partial Q_{La}^r}$ and $\partial_r^{\hat{a}} = \frac{\partial}{\partial Q_{R\hat{a}}^r}$. Recalling that $p_{L,a}{}^I$ and $p_{R,\hat{b}}{}^J$ are the left and right orthogonal projectors on the Grassmaniann $G_{p,q} = O(p,q)/[O(p) \times O(q)]$, one can use the effective derivation rules

$$\mathcal{D}_{a\hat{b}} p_{L,c}{}^I = \frac{1}{2} \delta_{ac} p_{R,\hat{b}}{}^I, \qquad\qquad \mathcal{D}_{a\hat{b}} p_{R,\hat{c}}{}^I = \frac{1}{2} \delta_{\hat{b}\hat{c}} p_{L,a}{}^I. \tag{3.44}$$

Acting with $\mathcal{D}_{e\hat{g}}$ on (2.32) we get

$$\mathcal{D}_{e\hat{g}} \Gamma^{(2)}_{\Lambda_{p,q}}[P] = \Gamma^{(2)}_{\Lambda_{p,q}} \Big[ \big( \mathcal{D}_{e\hat{g}} - 2\pi (Q_{Le} \Omega_2 Q_{R\hat{g}}) \big) P \Big], \tag{3.45}$$

where $(Q_{L,e} \Omega_2 Q_{R,\hat{g}}) = (\Omega_2)_{rs} Q_{La}^r Q_{R\hat{g}}^s$ is a short notation that will be used in the following.

It will prove useful to compute the commutation relations

$$[\Delta_2, \mathcal{D}_{e\hat{g}}] = 2(\partial_e \Omega_2^{-1} \partial_{\hat{g}}), \qquad\qquad [\Delta_2, Q_{Le}^r] = 2(\partial_e \Omega_2^{-1})^r, \tag{3.46}$$

$$[\Delta_2, Q_{Le}^r Q_{R\hat{g}}^s] = 2 Q_{Le}^r (\Omega_2^{-1} \partial_{\hat{g}})^s + 2 Q_{R\hat{g}}^s (\Omega_2^{-1} \partial_e)^r, \tag{3.47}$$

$$[\Delta_2, Q_{Le}^r Q_{Lf}^s] = 2 \delta_{ef} (\Omega_2^{-1})^{rs} + 4 Q_{L(e}^{(r} (\Omega_2^{-1} \partial_{f)})^{s)}, \tag{3.48}$$

with the Baker-Campbell-Hausdorff formula

$$e^{\frac{\Delta_2}{8\pi}}\mathcal{O}e^{-\frac{\Delta_2}{8\pi}} = \mathcal{O} + \frac{1}{8\pi}[\Delta_2,\mathcal{O}] + \frac{1}{2!}\frac{1}{(8\pi)^2}[\Delta_2,[\Delta_2,\mathcal{O}]] + \dots, \tag{3.49}$$

one obtains

$$\mathcal{D}_{e\hat{g}}\Gamma^{(2)}_{\Lambda_{p,q}}[P] = -2\pi\Gamma^{(2)}_{\Lambda_{p,q}}\Big[e^{-\frac{\Delta_2}{8\pi}}\big((Q_{Le}\Omega_2 Q_{R\hat{g}}) - \frac{1}{(4\pi)^2}(\partial_e\Omega_2^{-1}\partial_{\hat{g}})\big)e^{\frac{\Delta_2}{8\pi}}P\Big]. \tag{3.50}$$

Note that the derivation rules ensure that the constraints (3.16) are automatically satisfied at the level of the integrand, from the structure of (3.42) with $n = 0$. Antisymmetrizing (3.50) with $n = 0$, one obtains

$$\mathcal{D}_{[e}{}^{\hat{g}}\Gamma^{(2)}_{\Lambda_{p,q}}[P_{a_1...a_m,|b_1]...b_m}] = \Gamma^{(2)}_{\Lambda_{p,q}}\Big[e^{-\frac{\Delta_2}{8\pi}}\frac{1}{8\pi}(\partial_{[e}\Omega_2^{-1}\partial^{\hat{g}})e^{\frac{\Delta_2}{8\pi}}P_{a_1...a_m,|b_1]...b_m}\Big], \tag{3.51}$$

which vanishes identically since $e^{\frac{\Delta_2}{8\pi}}P_{a_1...a_m,b_1...b_m}$ does not depend on $Q_R$. The same argument goes for $\mathcal{D}_{[e}{}^{[\hat{e}}\mathcal{D}_f{}^{\hat{f}]}\Gamma^{(2)}_{\Lambda_{p,q}}[P_{a_1...a_m,b_1...b_m}]$ and $\mathcal{D}_{[e}{}^{[\hat{e}}\mathcal{D}_f{}^{\hat{f}}\mathcal{D}_g]{}^{\hat{g}]}\Gamma^{(2)}_{\Lambda_{p,q}}[P_{a_1...a_m,b_1...b_m}]$, and we conclude that for $m = 2$, the modular integral (3.39) satisfies

$$\mathcal{D}_{[e}{}^{\hat{e}}G_{a|b,|c],d} = 0, \qquad \mathcal{D}_{[e}{}^{\hat{e}}\mathcal{D}_f{}^{\hat{f}}G_{a]b,cd} = 0, \qquad \mathcal{D}_{[e}{}^{[\hat{e}}\mathcal{D}_f{}^{\hat{f}}\mathcal{D}_g]{}^{\hat{g}]}G_{ab,cd} = 0, \tag{3.52}$$

which thus establishes (3.16). Note that these properties are independent of the details of the function $1/\Phi_{k-2}(\Omega)$.

Now, the main equation (3.20) arises by applying the quadratic operator $\mathcal{D}^2_{ef} \equiv \mathcal{D}_{(e}{}^{\hat{g}}\mathcal{D}_{f)\hat{g}}$ on the lattice partition function with polynomial insertion, and commuting with the summation measure $e^{i\pi Q_L \Omega Q_L - i\pi Q_R \bar{\Omega} Q_R}$ of the partition function

$$\begin{aligned}
4\mathcal{D}^2_{ef}\Gamma^{(2)}_{\Lambda_{p,q}}[P] = \Gamma^{(2)}_{\Lambda_{p,q}}\Big[\Big(&4\mathcal{D}^2_{ef} - 8\pi(Q_{L(e}\Omega_2 Q_R{}^{\hat{g}})\mathcal{D}_{f)\hat{g}} - 2q\delta_{ef} \\
&+ 16\pi^2\,\mathrm{tr}\big[\Omega_2\big(Q_{Le}Q_{Lf} - \frac{\delta_{ef}}{4\pi}\Omega_2^{-1}\big)\Omega_2\big(Q_R^2 - \frac{q}{4\pi}\Omega_2^{-1}\big)\big]\Big)P\Big].
\end{aligned} \tag{3.53}$$

Using the commutation relations (3.46), one can re-express it to make modular invariance explicit

$$\begin{aligned}
4\mathcal{D}^2_{ef}\Gamma^{(2)}_{\Lambda_{p,q}}[P] = \Gamma^{(2)}_{\Lambda_{p,q}}\Big[&e^{-\frac{\Delta_2}{8\pi}}\Big(16\pi^2(Q_{Le}\Omega_2 Q_R{}^{\hat{g}})(Q_{R\hat{g}}\Omega_2 Q_{Lf}) - \delta_{ef}(qg + (Q_{R\hat{g}}\partial^{\hat{g}})) \\
&- q(Q_{L(e}\partial_{f)}) - 2(Q_{L(e}\Omega_2 Q_{R\hat{g}})(\partial^{\hat{g}}\Omega_2^{-1}\partial_f) + \frac{1}{16\pi^2}(\partial_e\Omega_2^{-1}\partial_{\hat{g}}\partial^{\hat{g}}\Omega_2^{-1}\partial_f)\Big)e^{\frac{\Delta_2}{8\pi}}P\Big],
\end{aligned} \tag{3.54}$$

and notice that all the terms in (3.54) except the first and last one will become linear tensorial combinations of the original partition function $\Gamma^{(2)}_{\Lambda_{p,q}}[P]$. The first term on the r.h.s of (3.54) can be rewritten as the action of the lowering operator for Siegel modular forms,

$$\bar{D}_{rs} = -\mathrm{i}\pi(\Omega_2(\Omega_2\partial_{\bar{\Omega}})^{\mathsf{T}})_{rs} = -\mathrm{i}\pi(\Omega_2)_{rt}(\Omega_2)_{su}\frac{\partial}{\partial\bar{\Omega}_{tu}}, \tag{3.55}$$

which take a weight $w$ representation $\mathrm{sym}^l$ modular form to a weight $w - 2$ representation $\mathrm{sym}^2 \otimes \mathrm{sym}^l$ modular form [62]. Indeed,

$$\begin{aligned}
\bar{D}_{rs}&\Gamma^{(2)}_{\Lambda_{p,q}}\Big[e^{-\frac{\Delta_2}{8\pi}}Q_{Le}^r Q_{Lf}^s e^{\frac{\Delta_2}{8\pi}}P\Big] \\
&= -\pi^2\Gamma^{(2)}_{\Lambda_{p,q}}\Big[\mathrm{tr}\big[\Omega_2\big(Q_R^2 - \frac{q}{4\pi}\Omega_2^{-1}\big)\Omega_2 e^{-\frac{\Delta_2}{8\pi}}(Q_{Le}Q_{Lf})e^{\frac{\Delta_2}{8\pi}}\big]P\Big] \\
&\qquad + \frac{1}{16}\Gamma^{(2)}_{\Lambda_{p,q}}\Big[(\partial_r^h\partial_{sh} + \partial_r^{\hat{h}}\partial_{s\hat{h}})e^{-\frac{\Delta_2}{8\pi}}Q_{Le}^r Q_{Lf}^s e^{\frac{\Delta_2}{8\pi}}P\Big] \\
&= \Gamma^{(2)}_{\Lambda_{p,q}}\Big[e^{-\frac{\Delta_2}{8\pi}}\big(\frac{1}{16}\partial_{rh}\partial_s^h Q_{Le}^r Q_{Lf}^s - \pi^2(Q_{Le}\Omega_2 Q_{R\hat{g}})(Q_R{}^{\hat{g}}\Omega_2 Q_{Lf}) \\
&\qquad - \frac{\pi}{2}\big((Q_{L(e}\Omega_2 Q_{R\hat{g}})(\partial^{\hat{g}}Q_{Lf})) - n(Q_{Le}\Omega_2 Q_{Lf})\big)e^{\frac{\Delta_2}{8\pi}}P\Big],
\end{aligned} \tag{3.56}$$

and the r.h.s. of (3.54) can thus be written as

$$
\begin{aligned}
4\mathcal{D}_{ef}^2 \Gamma_{\Lambda_{p,q}}^{(2)}\big[P\big] = \Gamma_{\Lambda_{p,q}}^{(2)}\Big[ & e^{-\frac{\Delta_2}{8\pi}}\Big((6-2q-Q_{R\hat{g}}\partial^{\hat{g}})\delta_{ef} + (6-q)(Q_{L(e}\partial_{f)}) \\
& + Q_{rLe}Q_{sLf}(\partial_L^2)^{rs} - 2(Q_{L(e}\Omega_2 Q_{R\hat{g}})(\partial^{\hat{g}}\Omega_2^{-1}\partial_f) \\
& - 8\pi\big((Q_{L(e}\Omega_2 Q_{R\hat{g}})(\partial^{\hat{g}}Q_{Lf)}) - n(Q_{Le}\Omega_2 Q_{Lf})\big) + \frac{1}{16\pi^2}(\partial_e\Omega_2^{-1}\partial_{\hat{g}}\partial^{\hat{g}}\Omega_2^{-1}\partial_f)\Big)e^{\frac{\Delta_2}{8\pi}}P\Big] \\
& - 16\bar{D}_{rs}\Gamma_{\Lambda_{p,q}}^{(2)}\Big[e^{-\frac{\Delta_2}{8\pi}}Q_{Le}^r Q_{Lf}^s e^{\frac{\Delta_2}{8\pi}}P\Big].
\end{aligned}
\tag{3.57}
$$

The third line contains contributions from partition functions with more or fewer momentum insertions, respectively, and the fourth line is to be computed explicitly. We now specialize to the case of interest and obtain

$$
\Delta_{ef}\Gamma_{\Lambda_{p,q}}^{(2)}\big[P_{ab,cd}\big] = -4\bar{D}_{rs}\Gamma_{\Lambda_{p,q}}^{(2)}\Big[e^{-\frac{\Delta_2}{8\pi}}Q_{Le}^r Q_{Lf}^s e^{\frac{\Delta_2}{8\pi}}P_{ab,cd}\Big],
\tag{3.58}
$$

where the operator $\Delta_{ef}$ is defined as

$$
\begin{aligned}
2\Delta_{ef}G_{ab,cd} \equiv &\, 2\mathcal{D}_{ef}^2 G_{ab,cd} + (q-3)\delta_{ef}G_{ab,cd} + (q-6)\big[\delta_{|e)(a}G_{b)(f|,cd} + \delta_{|e)(c}G_{d)(f|,ab}\big] \\
& - 3\delta_{\langle ab,}G_{cd\rangle,ef}.
\end{aligned}
\tag{3.59}
$$

Let us now return to the modular integral (3.39). In order to regularize the infrared divergences which arise when $q > 5$ (discussed in more detail in Appendix B.2.4), it is useful to first fold the integration domain $\Gamma_{2,0}(N)\backslash\mathcal{H}_2$ onto the fundamental domain $\mathcal{F}_2 = Sp(4,\mathbb{Z})\backslash\mathcal{H}_2$, and restrict the latter to truncated fundamental domain

$$
\mathcal{F}_{2,\Lambda,\eta} = \mathcal{F}_2 \cap \{\rho_2 \le \sigma_2 - v_2^2/\rho_2 \le \Lambda\} \cap \{|v| > \eta\}
\tag{3.60}
$$

excising both the non-separating degeneration at $\Omega_2 = i\infty$ and the separating degeneration at $v = 0$. We thus define

$$
G_{ab,cd}^{(p,q)}(\Lambda,\eta) = \int_{\mathcal{F}_{2,\Lambda,\eta}} \frac{\mathrm{d}^3\Omega_1 \mathrm{d}^3\Omega_2}{|\Omega_2|^3} \sum_{\gamma\in\Gamma_{2,0}(N)\backslash Sp(4,\mathbb{Z})} \left[\frac{\Gamma_{\Lambda_{p,q}}^{(2)}[P_{ab,cd}]}{\Phi_{k-2}(\Omega)}\right]_\gamma.
\tag{3.61}
$$

The renormalized integral (3.39) is defined as the limit of (3.61) as $\Lambda \to \infty$, $\eta \to 0$, possibly after subtracting divergent terms. Acting with the operator $\Delta_{ef}$ and using (3.58) one obtains

$$
\Delta_{ef}G_{ab,cd}^{(p,q)}(\Lambda,\eta) = -4\int_{\mathcal{F}_{2,\Lambda,\eta}} \frac{\mathrm{d}^3\Omega_1 \mathrm{d}^3\Omega_2}{|\Omega_2|^3} \sum_\gamma \left[\frac{1}{\Phi_{k-2}}\bar{D}_{rs}\Gamma_{\Lambda_{p,q}}^{(2)}\Big[e^{-\frac{\Delta_2}{8\pi}}Q_{Le}^r Q_{Lf}^s e^{\frac{\Delta_2}{8\pi}}P_{ab,cd}\Big]\right].
\tag{3.62}
$$

To compute the boundary term, we use Stokes' theorem in the form

$$
\int_{\partial\mathcal{F}_{2,\Lambda,\eta}^\Lambda} \frac{\mathrm{d}^5\Omega^{rs}}{|\Omega_2|^3}(\Omega_2)_{rt}(\Omega_2)_{su}(f^{tu}g) = \frac{2}{\pi}\int_{\mathcal{F}_{2,\Lambda,\eta}^\Lambda} \frac{\mathrm{d}^3\Omega_1 \mathrm{d}^3\Omega_2}{|\Omega_2|^3}\big(g\bar{D}_{rs}f^{rs} + f^{rs}\bar{D}_{rs}g\big),
\tag{3.63}
$$

where $f^{rs}$ and $g$ are modular form of $\Gamma_{2,0}(N)$ respectively of weight $w$ and representation $sym^2$, and weight $w' = 2 - w$ and trivial representation. The differential operator $\partial_{\bar{\Omega}}$ commutes with factors of $\Omega_2$ because of the natural connection $\bar{D}_{rs}$. Then, since $\bar{D}_{rs}1/\Phi_{k-2} = 0$ by holomorphicity, we obtain that the r.h.s. of (3.62) reads

$$
-2\pi\int_{\partial\mathcal{F}_{2,\Lambda,\eta}} \frac{\mathrm{d}^5\Omega^{rs}}{|\Omega_2|^3}(\Omega_2)_{rt}(\Omega_2)_{su}\sum_\gamma\left[\frac{1}{\Phi_{k-2}(\Omega)}\Gamma_{\Lambda_{p,q}}^{(2)}\Big[e^{-\frac{\Delta_2}{8\pi}}Q_{Le}^t Q_{Lf}^u e^{\frac{\Delta_2}{8\pi}}P_{ab,cd}\Big]\right]_\gamma.
\tag{3.64}
$$

The contributions from the $\Lambda$-dependent boundary of $\mathcal{F}_{2,\Lambda,\eta}$ lead to powerlike terms in $\Lambda$, which cancel in the renormalized integral, except for $q = 5$ or $q = 6$ where these divergent terms become logarithmic and are responsible for an anomalous term in the differential equation. These anomalous terms are computed in §B.2.5 and will be displayed in the final result below. Here we focus on the contribution from the boundary at $|v| = \eta$ due to the pole of the integrand at $v = 0$, which is cut-off independent for any $q$ and can be computed using Cauchy's theorem.

To compute the residue at $v = 0$, recall that the function $1/\Phi_{k-2}$ has a second order pole at $v = 0$ (cf. (A.44)) and behaves as $\Phi_{k-2} \sim (2\pi i v)^2 \Delta_k(\rho) \times \Delta_k(\sigma) + O(v^4)$. The only cosets $\gamma$ preserving the pole at $v = 0$ are those in $\gamma \in (\Gamma_0(N) \backslash SL(2,\mathbb{Z}))_\rho \times (\Gamma_0(N) \backslash SL(2,\mathbb{Z}))_\sigma$. Adding up these contributions, we find that the residue of the integrand at $v = 0$ is

$$\frac{1}{\rho_2^2 \sigma_2^2} \frac{i}{\Delta_k(\rho)\Delta_k(\sigma)} \sum_{\substack{\gamma \in (\Gamma_0(N)\backslash SL(2,\mathbb{Z}))_\rho \\ \times (\Gamma_0(N)\backslash SL(2,\mathbb{Z}))_\sigma}} \Gamma_{\Lambda_{p,q}}^{(2)} \left[ e^{-\frac{\Delta_2}{8\pi}} Q_{Lk}^1 Q_L^{2k} Q_{Le}^1 Q_{Lf}^2 e^{\frac{\Delta_2}{8\pi}} P_{ab,cd} \right]\Big|_\gamma \left( \Omega = \begin{pmatrix} \rho & 0 \\ 0 & \sigma \end{pmatrix} \right). \tag{3.65}$$

Near the boundary at $|v| = \eta$, the fundamental domain $\mathcal{F}_{2,\Lambda,\eta}$ reduces to $\mathcal{F}_1(\rho) \times \mathcal{F}_1(\sigma) \times \{|v| > \eta\}/\mathbb{Z}_2 \times Z_2$ where the first $\mathbb{Z}_2$ exchanges $\rho$ and $\sigma$ while the second sends $v \mapsto -v$. Thus, the sum in (3.65) factorizes into two genus-one integrands, leading to

$$\begin{aligned}\Delta_{ef} G_{ab,cd}^{(p,q)}(\Lambda,\eta) &= -\pi \big( F_{abk(e}^{(p,q)}(\Lambda) F_{f)cd}^{(p,q)\ k}(\Lambda) - F_{ak|c)(e}^{(p,q)}(\Lambda) F_{f)(d|b}^{(p,q)\ k}(\Lambda) \big) + \ldots \\ &= -\frac{3\pi}{2} F_{|e)k\langle ab,}^{(p,q)}(\Lambda) F_{cd\rangle}^{(p,q)k}{}_{(f|}(\Lambda) + \ldots, \end{aligned} \tag{3.66}$$

where the dots denote contributions from the $\Lambda$-dependent boundary, discussed in detail in Appendix B.2.4, while $F_{abcd}^{(p,q)}(\Lambda)$ is the genus-one regularized modular integral

$$F_{abcd}^{(p,q)}(\Lambda) = \int_{\mathcal{F}_{1,\Lambda}} \frac{d\rho_1 d\rho_2}{\rho_2^2} \sum_{\gamma \in \Gamma_0(N)\backslash SL(2,\mathbb{Z})} \left[ \frac{1}{\Delta_k} \Gamma_{\Lambda_{p,q}} \big[ P_{abcd} \big] \right]_\gamma. \tag{3.67}$$

This establishes (3.20) with $\varpi = \pi$. We show in Appendix B.2.5 that the divergent terms from the $\Lambda$-dependent boundary of $\mathcal{F}_{2,\Lambda,\eta}$ combine consistently such that the renormalised coupling satisfies the same differential equation (3.20), but for $q = 5$ or $q = 6$, for which one gets additional linear source terms. For the perturbative string amplitude, $v = N$, the additional source term vanishes for $q = 5$, and for $q = 6$ it can be ascribed to the mixing between the analytic and the non-analytic parts of the amplitude. In this case one obtains (B.96)

$$\Delta_{ef} G_{ab,cd}^{(p,q)} = -\frac{3\pi}{2} F_{|e)k\langle ab,}^{(p,q)} F_{cd\rangle}^{(p,q)k}{}_{(f|} - \delta_{q,6} \frac{3}{16\pi} \big( \delta_{ef} \delta_{\langle ab,} + 2\delta_{e\langle (a} \delta_{b),|f|} \big) F_{cd\rangle k}^{(p,q)\ k}, \tag{3.68}$$

where $\Delta_{ef}$ was defined in (3.59).

# 4 Weak coupling expansion of exact $\nabla^2(\nabla\phi)^4$ couplings

In this section, we study the asymptotic expansion of the proposal (1.5) in the limit where the heterotic string coupling $g_3$ goes to zero, and show that it reproduces the known tree-level and one-loop amplitudes, along with an infinite series of NS5-brane, Kaluza–Klein monopole and H-monopole instanton corrections. For the sake of generality, we analyze the family of modular integrals (2.30), which we copy again for convenience,

$$G_{ab,cd}^{(p,q)} = \text{R.N.} \int_{\Gamma_{2,0}(N)\backslash\mathcal{H}_2} \frac{d^3\Omega_1 d^3\Omega_2}{|\Omega_2|^3} \frac{\Gamma_{\Lambda_{p,q}}^{(2)} [P_{ab,cd}]}{\Phi_{k-2}(\Omega)}, \tag{4.1}$$

for a level $N$ even lattice $\Lambda_{p,q}$ of arbitrary signature $(p,q)$, in the limit near the cusp where $O(p,q)$ is broken to $O(1,1) \times O(p-1,q-1)$, so that the moduli space decomposes into

$$G_{p,q} \to \mathbb{R}^+ \times G_{p-1,q-1} \ltimes \mathbb{R}^{p+q-2} \,. \tag{4.2}$$

For simplicity, we first discuss the maximal rank case $N = 1$, $p - q = 16$, where the integrand is invariant under the full Siegel modular group $Sp(4, \mathbb{Z})$, before dealing with the case of $N$ prime, where the integrand is invariant under the congruence subgroup $\Gamma_{2,0}(N)$.

The reader uninterested by the details of the derivation may skip ahead to §4.3, where we specialize to the values $(p,q) = (2k, 8)$ relevant for the $\nabla^2(\nabla\phi)^4$ couplings in $D = 3$ and interpret the various contributions as perturbative and non-perturbative effects in heterotic string theory compactified on $T^7$. In §6.4 we apply the computation in this section to the case $(p,q) = (21, 5)$ relevant for $\nabla^2\mathcal{H}^4$ couplings in type IIB string theory compactified on $K3$.

## 4.1 $O(p,q) \to O(p-1,q-1)$ for even self-dual lattices

In this subsection we assume that the lattice $\Lambda_{p,q}$ is even self-dual and factorizes in the limit (4.2) as

$$\Lambda_{p,q} \to \Lambda_{p-1,q-1} \oplus \mathbb{I}_{1,1} \,. \tag{4.3}$$

We shall denote by $R$ the coordinate on $\mathbb{R}^+$, $\varphi$ the coordinates on $G_{p-1,q-1}$ and by $a^I$, $I = 1 \ldots p + q - 2$ the coordinates on $\mathbb{R}^{p+q-2}$. The variable $R > 0$ parametrizes a one-parameter subgroup $e^{RH_0}$ in $O(p,q)$, such that the action of the non-compact Cartan generator $H_0$ on the Lie algebra $\mathfrak{so}_{p,q}$ decomposes into

$$\mathfrak{so}_{p,q} \simeq (\mathbf{p+q-2})^{(-2)} \oplus (\mathfrak{gl}_1 \oplus \mathfrak{so}_{p-1,q-1})^{(0)} \oplus (\mathbf{p+q-2})^{(2)} \,, \tag{4.4}$$

while the coordinates $a^I$ parametrize the unipotent subgroup obtained by exponentiating the grade 2 component in this decomposition.

The lattice vectors are now labelled according to the choice of A-cycle on the genus-two Riemann surface. They thus take value take value in double copy of the original lattice $\Lambda_{p,q} \oplus \Lambda_{p,q}$. Thus, the generic charge vector $(Q_{1\mathcal{I}}, Q_{2\mathcal{I}}) \in \Lambda_{p,q} \oplus \Lambda_{p,q} \simeq \mathbf{2}^{(-2)} \oplus (\mathbf{2} \otimes (\mathbf{p+q-2}))^{(0)} \oplus \mathbf{2}^{(2)},$[9] decomposes into

$$(Q_{\mathcal{I}}^1, Q_{\mathcal{I}}^2) = (n^1, n^2, \widetilde{Q}_I^1, \widetilde{Q}_I^2, m^1, m^2), \tag{4.5}$$

where $(n^1, n^2, m^1, m^2) \in \mathbb{I}_{1,1} \oplus \mathbb{I}_{1,1}$ and $(\widetilde{Q}_{\mathcal{I}}^1, \widetilde{Q}_{\mathcal{I}}^2) \in \Lambda_{p-1,q-1} \oplus \Lambda_{p-1,q-1}$, such that $Q^r \cdot Q^r = -2m^r n^r + \widetilde{Q}^r \widetilde{Q}^r$ (with no summation on $r$). The orthogonal projectors defined by $Q_L^r \equiv p_L^I Q_I^r$ and $Q_R^r \equiv p_R^I Q_I^r$ decompose according to

$$
\begin{aligned}
p_{L,1}^{\mathcal{I}} Q_{\mathcal{I}}^r &= \frac{1}{R\sqrt{2}} \left( m^r + a \cdot \widetilde{Q}^r + \frac{1}{2} a \cdot a\, n^r \right) - \frac{R}{\sqrt{2}} n^r \,, \\
p_{L,a}^{\mathcal{I}} Q_{\mathcal{I}}^r &= \widetilde{p}_{L,a}^I (\widetilde{Q}_I^r + n^r a_I) \,, \\
p_{R,1}^{\mathcal{I}} Q_{\mathcal{I}}^r &= \frac{1}{R\sqrt{2}} \left( m^r + a \cdot \widetilde{Q}^r + \frac{1}{2} a \cdot a\, n^r \right) + \frac{R}{\sqrt{2}} n^r \,, \\
p_{R,\hat{a}}^{\mathcal{I}} Q_{\mathcal{I}}^r &= \widetilde{p}_{R,\hat{a}}^I (\widetilde{Q}_I^r + n^r a_I) \,,
\end{aligned}
\tag{4.6}
$$

where $\widetilde{p}_{L,a}^I, \widetilde{p}_{R,\hat{a}}^I$ ($a = 2 \ldots q + 16$, $\hat{a} = 2 \ldots q$) are orthogonal projectors in $G_{p-1,q-1}$ satisfying $\widetilde{Q}^r \cdot \widetilde{Q}^s = \widetilde{Q}_L^r \cdot \widetilde{Q}_L^s - \widetilde{Q}_R^r \widetilde{Q}_R^s$. In the following, we shall denote $|\widetilde{Q}_R^r| \equiv \sqrt{\widetilde{p}_{R,\hat{a}}^I \widetilde{p}_R^{J\hat{a}} \widetilde{Q}_I^r \widetilde{Q}_J^r}$.

To study the behavior of (4.1) in the limit $R \gg 1$, it is useful to perform a Poisson resummation on the momenta $(m_1, m_2)$. For a lattice partition function $\Gamma_{\Lambda_{p,q}}^{(2)}$ with or without insertion,

---

[9]We use $\mathcal{I}$ to label indices from 1 to $p+q$ in this paragraph to differentiate them from the indices on the sublattice.

we must distinguish whether the indices lie along the direction 1 or along the directions $\alpha$. The result can be obtain by applying the corresponding derivative polynomial with respect to $(y_{r,1}, y_{r,\alpha})$ to the following partition function

$$
\Gamma^{(2)}_{\Lambda_{p,q}}\left[e^{2\pi i y_a \cdot \widetilde{Q}^a + \frac{\pi}{2} y_a \cdot \Omega_2^{-1} \cdot y^a}\right] =
$$
$$
R^2 \sum_{(\mathbf{n},\mathbf{m}) \in \mathbb{Z}^4} e^{-\pi R^2 (\mathbf{n}\,\mathbf{m})\binom{\Omega}{\mathbb{1}} \cdot \Omega_2^{-1} \cdot \left[(\mathbf{n}\,\mathbf{m})\binom{\bar{\Omega}}{\mathbb{1}}\right]^\intercal} e^{\frac{2\pi R}{\sqrt{2}} y_1 \cdot \Omega_2^{-1} \cdot \left[(\mathbf{n}\,\mathbf{m})\binom{\bar{\Omega}}{\mathbb{1}}\right]^\intercal}
$$
$$
\times \Gamma^{(2)}_{\Lambda_{p-1,q-1}}\left[e^{2\pi i \mathbf{m}^\intercal \cdot (a^I \widetilde{Q}_I + \frac{1}{2} a^I a_I \mathbf{n})} e^{2\pi i y_\alpha \cdot \widetilde{Q}^\alpha + \frac{\pi}{2} y_\alpha \cdot \Omega_2^{-1} \cdot y^\alpha}\right], \quad (4.7)
$$

where we denote the winding and momenta doublets $\mathbf{n} = (n_1, n_2)$, $\mathbf{m} = (m_1, m_2)$, and we use Einstein summation convention for indices $I = 1, \ldots, p+q-2$ and $\alpha = 2, \ldots, p$. In this representation, modular invariance is manifest, since a transformation $\Omega \mapsto (A\Omega + B)(C\Omega + D)^{-1}$ (A.2) can be compensated by a linear transformation $(\mathbf{n}, \mathbf{m}) \mapsto (\mathbf{n}, \mathbf{m})\left(\begin{smallmatrix} D^\intercal & -B^\intercal \\ -C^\intercal & D^\intercal \end{smallmatrix}\right)$, $y_1 \mapsto y_1 \cdot (C\Omega + D)$, under which the third line of (4.7) transforms as a weight $\frac{p-q}{2}$ modular form.

We can therefore compute the integral using the orbit method [63–67], namely decompose the sum over $(\mathbf{n}, \mathbf{m})$ into various orbits under $Sp(4, \mathbb{Z})$, and for each orbit $\mathcal{O}$, retain the contribution of a particular element $\varsigma \in \mathcal{O}$ at the expense of extending the integration domain $\mathcal{F}_2 = Sp(4, \mathbb{Z}) \backslash \mathcal{H}_2$ to $\Gamma_\varsigma \backslash \mathcal{H}_2$, where $\Gamma_\varsigma$ is the stabilizer of $\varsigma$ in $Sp(4, \mathbb{Z})$. The integration domain is unfolded according to the formula

$$
\bigcup_{\gamma \in \Gamma_\varsigma \backslash Sp(4,\mathbb{Z})} \gamma \cdot \mathcal{F}_2 = \Gamma_\varsigma \backslash \mathcal{H}_2 , \quad (4.8)
$$

where one must take into account that $-\mathbb{1} \in Sp(4, \mathbb{Z})$ acts trivially on $\mathcal{H}_2$. The coset representative $\varsigma \in \mathcal{O}$, albeit arbitrary, is usually chosen so as to make the unfolded domain $\Gamma_\varsigma \backslash \mathcal{H}_2$ as simple as possible. In the present case, there are two types of orbits:

**The trivial orbit** $(\mathbf{n}, \mathbf{m}) = (0, 0, 0, 0)$ produces, up to a factor of $R^2$, the integrals (4.1) for the lattice $\Lambda_{p-1,q-1}$, provided none of the indices $ab, cd$ lie along the direction 1,

$$
G^{(p,q),0}_{\alpha\beta,\gamma\delta} = R^2 \, G^{(p-1,q-1)}_{\alpha\beta,\gamma\delta} , \quad (4.9)
$$

while it vanishes otherwise.

**The rank-one orbits** correspond to terms with $(\mathbf{n}, \mathbf{m}) \neq (0, 0, 0, 0)$. Setting $(n_1, n_2, m_1, m_2) = k(c_3, c_4, d_3, d_4)$, with $\gcd(c_3, c_4, d_3, d_4) = 1$ and $k \neq 0$, the quadruplet $(c_3, c_4, d_3, d_4)$ can always be rotated by an element of $Sp(4, \mathbb{Z})$ into $(0, 0, 0, 1)$, whose stabilizer inside $Sp(4, \mathbb{Z})$ is $\Gamma^J_1$ (4.10)

$$
\Gamma^J_1 = \left\{ \begin{pmatrix} a & 0 & b & \mu' \\ \lambda & 1 & \mu & \kappa \\ c & 0 & d & -\lambda' \\ 0 & 0 & 0 & 1 \end{pmatrix}, (\lambda, \mu) = (\lambda', \mu')\begin{pmatrix} a & b \\ c & d \end{pmatrix}, \begin{pmatrix} a & b \\ c & d \end{pmatrix} \in SL(2, \mathbb{Z}), (\kappa, \lambda, \mu) \in \mathbb{Z}^3 \right\} ,
$$
$$
(4.10)
$$

which is a central extension of the Jacobi group $SL(2, \mathbb{Z}) \ltimes \mathbb{Z}^2$ in which the triple $(\kappa, \lambda, \mu) \in \mathbb{Z}^3$ parametrizes the Heisenberg group $H_{2,1}(\mathbb{Z})$.[10]

---

[10] They satisfy the group multiplication law $(\lambda, \mu, \kappa) \cdot (\lambda', \mu', \kappa') = (\lambda + \lambda', \mu + \mu', \kappa + \kappa' + \lambda\mu' - \lambda'\mu)$.

Thus, quadruplets $(c_3, c_4, d_3, d_4)$ with $\gcd(c_3, c_4, d_3, d_4) = 1$ are in one-to-one correspondence with elements of $\Gamma_1^J \backslash Sp(4, \mathbb{Z})$. For each $k \in \mathbb{Z}$, one can therefore unfold the integration domain $Sp(4, \mathbb{Z}) \backslash \mathcal{H}_2$ to

$$\Gamma_1^J \backslash \mathcal{H}_2 = \mathbb{R}_t^+ \times (SL(2, \mathbb{Z}) \backslash \mathcal{H}_1)_\rho \times \left( (\mathbb{R}/\mathbb{Z})^3 / \mathbb{Z}_2 \right)_{u_1, u_2, \sigma_1}, \tag{4.11}$$

provided one keeps only the term $(c_3, c_4, d_3, d_4) = (0, 0, 0, 1)$ in the sum, and where $\mathbb{Z}_2$ comes from the element $-\mathbb{1} \in SL(2, \mathbb{Z})$ leaving $\rho$ invariant but acting as $(u_1, u_2) \to (-u_1, -u_2)$. In practice, we integrate $u_1, u_2$ over $\mathbb{R}/\mathbb{Z}$ and multiply the integral by a factor $1/2$. We parametrize the domain $\Gamma_1^J \backslash \mathcal{H}_2$ by $t = \frac{|\Omega_2|}{\rho_2}$, $\rho$, and $(u_1, u_2, \sigma_1) = (v_1 - v_2 \rho_1/\rho_2, v_2/\rho_2, \sigma_1)$.

The resulting contribution can be expressed in terms of the $y$ variables (4.7). Changing $y_{ra}$ variables as $(y'_{11}, y'_{21}, y'_{1\alpha}, y'_{2\alpha}) = (y_{11}, y_{11} u_2 - y_{21}, y_{1\alpha}, y_{1\alpha} u_2 - y_{2\alpha})$, we obtain

$$G_{ab,cd}^{(p,q),1} = \frac{R^2}{2} \int_0^\infty \frac{dt}{t^3} \int_{(\mathbb{R}/\mathbb{Z})^3} du_1 du_2 d\sigma_1 \int_{\mathcal{F}_1} \frac{d\rho_1 d\rho_2}{\rho_2^2} \frac{\mathcal{P}_{ab,cd}(\frac{\partial}{\partial y'})}{\Phi_{10}} \sum_{k \neq 0} e^{-\frac{\pi R^2 k^2}{t}} \Gamma_{\Lambda_{p-1,q-1}}^{(2)} \Big[ e^{2\pi i k a^I \widetilde{Q}_{2I}}$$
$$\times \exp\Big( 2\pi \Big( \frac{R}{\sqrt{2}} \frac{k}{t} y'_{21} + i y'_{1\alpha} (Q_L^{1\alpha} + u_2 Q_L^{2\alpha}) - i y'_{2\alpha} Q_L^{2\alpha} + \frac{1}{4\rho_2} y'_{1\alpha} y_1'^{\alpha} + \frac{1}{4t} y'_{2\alpha} y_2'^{\alpha} \Big) \Big) \Big], \tag{4.12}$$

where

$$\mathcal{P}_{ab,cd}\Big( \frac{\partial}{\partial y} \Big) = \varepsilon_{rt} \varepsilon_{su} \frac{1}{(2\pi i)^4} \frac{\partial}{\partial y_r^{(a}} \frac{\partial}{\partial y_s^{b)}} \frac{\partial}{\partial y_t^{(c}} \frac{\partial}{\partial y_u^{d)}}. \tag{4.13}$$

The integral over $\Gamma_1^J \backslash \mathcal{H}_2$ can be computed by inserting the Fourier–Jacobi expansion

$$\frac{1}{\Phi_{10}} = \sum_{\substack{m \in \mathbb{Z} \\ m \geq -1}} \psi_m(\rho, v) q^m. \tag{4.14}$$

The integral over $\sigma_1$ picks up the Jacobi form $\psi_m(\rho, v)$ with $m = -\frac{1}{2} \widetilde{Q}_2^2$.

For $\widetilde{Q}_2 = 0$, one has from (A.54), $\psi_0 = c(0) \mathcal{P}/\Delta$ where here $\mathcal{P}$ denotes the (rescaled) Weierstrass function (A.55) and $c(0) = 24$ is the zero-th Fourier coefficient in $1/\Delta = \sum_{m \geq -1} c(m) q^m$. The integral over $\sigma_1$ is trivial while the integral over $u_1, u_2$ is computed using (A.72),

$$\int_{-1/2}^{1/2} du_1 \int_{-1/2}^{1/2} du_2 \, \psi_0(\rho, u_1 + \rho u_2) = \frac{c(0) \widehat{E}_2}{12 \Delta}, \tag{4.15}$$

where $\widehat{E}_2(\rho) = E_2(\rho) - \frac{3}{\pi \rho_2}$ is the non-holomorphic completion of the weight 2 Eisenstein series. The contributions with $\widetilde{Q}_2 = 0$ therefore lead to the integral (after exchanging the order of sum and integral)

$$G_{ab,cd}^{(p,q),1} = R^2 \frac{c(0)}{24} \sum_{k \neq 0} \int_0^\infty \frac{dt}{t} t^{\frac{q-5}{2}} e^{-\frac{\pi R^2 k^2}{t}} \mathcal{P}_{ab,cd}\Big( \frac{\partial}{\partial y'} \Big) e^{2\pi i \left( \frac{R}{i\sqrt{2}} \frac{k}{t} y'_{21} + \frac{1}{4it} y'_{2\alpha} y_2'^{\alpha} \right)}$$
$$\times \int_{\mathcal{F}_1} \frac{d\rho_1 d\rho_2}{\rho_2^2} \frac{\widehat{E}_2}{\Delta(\rho)} \Gamma_{\Lambda_{p-1,q-1}} \Big[ e^{2\pi i \left( y'_{1\alpha} Q_L^{1\alpha} + \frac{1}{4i\rho_2} y'_{1\alpha} y_1'^{\alpha} \right)} \Big], \tag{4.16}$$

leading to the constant terms in the Fourier expansion of $G_{ab,cd}^{(p,q)}$

$$G_{\alpha\beta,\gamma\delta}^{(p,q),1,0} = -R^{q-5} \xi(q-6) \frac{c(0)}{16\pi} \delta_{\langle \alpha\beta,} G_{\gamma\delta \rangle}^{(p-1,q-1)},$$
$$G_{\alpha\beta,11}^{(p,q),1,0} = -R^{q-5} \xi(q-6)(7-q) \frac{c(0)}{48\pi} G_{\alpha\beta}^{(p-1,q-1)}, \tag{4.17}$$

and $G^{(p,q),1,0}_{\alpha\beta,\gamma1} = 0$. Note that they are the only components by symmetry of the indices $ab, cd$. Here $G^{(p,q)}_{ab}$ is the genus-one modular integral defined in (2.29) with $N = 1$ and $\xi(s) = \pi^{-s/2}\Gamma(s/2)\zeta(s) = \xi(1-s)$ is the completed Riemann zeta function.

**The missing constant term:**   It is clear from the differential equation (3.20) that (4.17) does not give all the power-like terms: indeed, the coupling $F^{(p,q)}$ appearing on the r.h.s. of (3.20) behaves schematically in the same limit as [22, (4.37)]

$$F^{(p,q)} \sim R F^{(p-1,q-1)} + \xi(q-6)R^{q-6} + \mathcal{O}(e^{-R}) . \tag{4.18}$$

The power-like terms (4.17) can be checked to satisfy the differential constraint with the source term $R^{q-5}F^{(p-1,q-1)}$ appearing in the square of $F^{(p,q)}$, but the accompanying source term $\xi(q-6)^2R^{2q-12}$ requires that $G^{(p,q)}_{ab,cd}$ should also include a term proportional to $R^{2q-12}$. We shall now argue that these terms originate from the intersection of the separating and non-separating degenerations described by the figure-eight supergravity diagram depicted in Figure 1ii). In the region $|\Omega_2| \gg 1$, the fundamental domain asymptotes to the domain $\mathcal{P}_2/GL(2,\mathbb{Z}) \times [0,1]^3$, where $\Omega_2$ parametrizes the first factor. In the case where all external indices are along the subgrassmaniann, the dominant contributions in this limit have $\widetilde{Q}_1 = \widetilde{Q}_2 = 0$ and vanishing winding number $(n_1, n_2)$ along the circle. The sum over dual momenta $(m_1, m_2)$ running in the two loops leads to

$$\frac{3}{16\pi^2}\delta_{\langle\alpha\beta},\delta_{\gamma\delta\rangle}R^2 \int_{\frac{\mathcal{P}_2}{GL(2,\mathbb{Z})}\times[0,1]^3} \frac{d^3\Omega_1 d^3\Omega_2}{|\Omega_2|^{4-\frac{q-1}{2}}} \frac{\sum_{m_r\in\mathbb{Z}^2} e^{-\pi R^2 m_r[\Omega_2^{-1}]^{rs}m_s}}{\Phi_{10}} . \tag{4.19}$$

Using (A.90), the integral over $\Omega_1$ leads to a delta function supported at $\nu_2 = 0$ and its images under the action of $GL(2,\mathbb{Z})$ (modulo the center). After unfolding, the remaining integral then factorizes into two integrals over $\rho_2$ and $\sigma_2$. Assuming that this contribution is accurately computed by this integral by extending the integration domain of $\rho_2$ and $\sigma_2$ to $\mathbb{R}^+$, one obtains the correct power-like term

$$
\begin{aligned}
G^{(p,q),1,0'}_{\alpha\beta,\gamma\delta} &= -\frac{3}{64\pi^3}R^{2q-12}\left[\xi(q-6)c(0)\right]^2\delta_{\langle\alpha\beta},\delta_{\gamma\delta\rangle} , \\
G^{(p,q),1,0'}_{\alpha\beta,11} &= -\frac{1}{32\pi^3}R^{2q-12}\left[\xi(q-6)c(0)\right]^2(7-q)\delta_{\alpha\beta} ,
\end{aligned}
\tag{4.20}
$$

where the second line — the other non-vanishing polarization — can be deduced in a similar fashion. While the power-like terms (4.20) are not captured by the unfolding trick in the degeneration $(p,q) \to (p-1,q-1)$, we shall be able to recover them below from the degeneration $(p,q) \to (p-2,q-2)$, see (5.26).

The fact that the unfolding method does not give the full result is seemingly due to the non-absolute convergence of the integral near the separating locus. In principle, the missing contributions can be determined by checking the differential equation (3.20). In Appendix E.4 we derive the contributions (4.20) rigorously in this fashion. The same analysis also implies that there exists additional exponentially suppressed corrections to the constant term due to instanton–anti-instanton contributions. For what concerns non-trivial Fourier coefficients, we shall argue in §5.1 (and specifically in Appendix E.1) that the unfolding method is in fact reliable.

**Exponentially suppressed corrections:**   Contributions from non-zero vectors $\widetilde{Q}_2$ lead to exponentially suppressed contributions, which depend on the axions through a phase factor $e^{2\pi ika^I\widetilde{Q}_{2I}}$. Each Jacobi form $\psi_m(\rho,\nu)$ in (4.14) can be decomposed as the sum of a finite

and polar contributions, $\psi_m = \widehat{\psi}_m^P + \widehat{\psi}_m^F$ (see §A.5), where $\widehat{\psi}_m^F$ is an almost holomorphic Jacobi form, and $\widehat{\psi}_m^P$ is proportional to a completed non-holomorphic Appell–Lerch sum. For $m = -1$, the finite part vanishes and the polar part requires special treatment. In either case, the integral over $\sigma_1$ enforces $\widetilde{Q}_2^2 = -2m$.

We first treat the finite contributions $\widehat{\psi}_m^F(\rho, \nu)$ with $m \geq 0$ according to whether $\widetilde{Q}_2^2 = 0$ or $\widetilde{Q}_2^2 \neq 0$, and then consider the polar contributions:

1. In the case $\widetilde{Q}_2^2 = 0$, since $\widehat{\psi}_0^F = \frac{c(0)\widehat{E}_2}{12\Delta}$ and does not depend on $\nu$, the integral over $u_1$ receives only contributions from vectors $\widetilde{Q}_1$ such that $\widetilde{Q}_1 \cdot \widetilde{Q}_2 = 0$. To express the remaining sum, we choose a second null vector $\widetilde{Q}_2'$ such that $(\widetilde{Q}_2, \widetilde{Q}_2') = m_2$, where $m_2$, which we also denote by $\gcd(\widetilde{Q}_2)$, is the largest integer such that $\frac{1}{m_2}\widetilde{Q}_2 \in \Lambda_{p-1,q-1}$. The vectors $\widetilde{Q}_1$ orthogonal to $\widetilde{Q}_2$ are then of the form $\widetilde{Q}_1 = \widetilde{Q}_1^\perp + \frac{m_1}{m_2}\widetilde{Q}_2$ where $\widetilde{Q}_1^\perp$ is orthogonal to both $\widetilde{Q}_2$ and $\widetilde{Q}_2'$. We denote the resulting lattice by $\Lambda_{p-2,q-2}$. This parametrization is not unique, but the result of the integral will be independent of the choice of $\widetilde{Q}_2'$, in other words it is a function of the Levi subgroup of the stabilizer of $\widetilde{Q}_2$ inside $O(p-1, q-1)$. The sum over $\widetilde{Q}_1$ therefore becomes a sum over $\widetilde{Q}_1^\perp \in \Lambda_{p-2,q-2}$ and $m_1 = m_2 s + r$, $s \in \mathbb{Z}$, $r \in \mathbb{Z}_{m_2}$. The sum over $s$ can be used to unfold the integral over $u_2 \in [-\frac{1}{2}, \frac{1}{2}]$ to the full $\mathbb{R}$ axis, as one can see from (4.12), while the dependence on $r$ can be absorbed by a translation in $u_2$ and therefore leads to an overall factor $m_2$. The integral thus becomes, for a given null vector $\widetilde{Q}_2$,

$$
\begin{aligned}
R^2 \frac{m_2}{2} \sum_{k \neq 0} e^{2\pi i k \widetilde{Q}_{2I} a^I} &\int_{\mathbb{R}^+} \frac{dt}{t} t^{\frac{q-5}{2}} e^{-\frac{\pi R^2 k^2}{t} - 2\pi t |\widetilde{Q}_{2R}|^2} \int_{\mathcal{F}_1} \frac{d\rho_1 d\rho_2}{\rho_2^2} \rho_2^{\frac{q-1}{2}} \frac{c(0)\widehat{E}_2}{12\Delta} \\
&\times \int_{\mathbb{R}} du_2 \sum_{\widetilde{Q}_1^\perp \in \Lambda_{p-2,q-2}} q^{\frac{1}{2}(\widetilde{Q}_1^\perp + u_2 \widetilde{Q}_2)_L^2} \bar{q}^{\frac{1}{2}(\widetilde{Q}_1^\perp + u_2 \widetilde{Q}_2)_R^2} \\
&\times \mathcal{P}_{ab,cd}\left(\frac{\partial}{\partial y'}\right) e^{2\pi i \left(\frac{R}{i\sqrt{2}}\frac{k}{t}y_{21}' + y_{1\alpha}'(\widetilde{Q}_1^\perp + u_2 \widetilde{Q}_2)_L^\alpha - y_{2\alpha}' \widetilde{Q}_{2L}^\alpha + \frac{1}{4i\rho_2} y_{1\alpha}' y_1'^\alpha + \frac{1}{4it} y_{2\alpha}' y_2'^\alpha\right)}\bigg|_{y'=0}.
\end{aligned}
\tag{4.21}
$$

The Gaussian integral over $u_2$ removes the dependence on the unipotent part of the stabilizer of $Q = k\widetilde{Q}_2$, leaving a modular integral of a genus-one partition function $G_{\alpha\beta,0}^{(p-1,q-1)\perp}$ for the lattice $\Lambda_{p-2,q-2}$ depending only on the sub-Grassmaniann $G_{p-2,q-2} \subset G_{p-1,q-1}$ parametrizing the Levi component of this stabilizer, given by

$$
\begin{aligned}
G_{F,\alpha\beta,0}^{(p-1,q-1)\perp}(Q) = \frac{\gcd(Q)}{12} &\int_{\mathcal{F}_1} \frac{d\rho_1 d\rho_2}{\rho_2^2} \frac{\widehat{E}_2}{\Delta(\rho)} \rho_2^{\frac{q-2}{2}} \sum_{\widetilde{Q} \in \Lambda_{p-2,q-2}} q^{\frac{1}{2}\widetilde{Q}_L^2} \bar{q}^{\frac{1}{2}\widetilde{Q}_R^2} e^{2\pi\rho_2 \frac{(\widetilde{Q}_R \cdot Q_R)^2}{Q_R^2}} \\
&\times \left[\left(\widetilde{Q}_{L\alpha} - \frac{\widetilde{Q}_R \cdot Q_R}{Q_R^2} Q_{L\alpha}\right)\left(\widetilde{Q}_{L\beta} - \frac{\widetilde{Q}_R \cdot Q_R}{Q_R^2} Q_{L\beta}\right) - \frac{1}{4\pi\rho_2}\left(\delta_{\alpha\beta} - \frac{Q_{L\alpha} Q_{L\beta}}{Q_R^2}\right)\right],
\end{aligned}
\tag{4.22}
$$

where we write $\widetilde{Q}_1$ as $\widetilde{Q}$ for simplicity. Note that the integrand only depends on $\widetilde{Q}$ through $\widetilde{Q} - Q\frac{\widetilde{Q}_R \cdot Q_R}{Q_R^2}$, and so is invariant under $\widetilde{Q} \to \widetilde{Q} + \epsilon Q$ for any $\epsilon \in \mathbb{R}$ such that the sum is defined on the quotient lattice $\Lambda_{p-1,q-1} \mathrm{mod} \frac{Q}{\gcd(Q)}$ with the constraint $Q \cdot \widetilde{Q} = 0$, and does not depend on the specific choice of $\Lambda_{p-2,q-2}$.

We find that the Fourier coefficient with charge $Q \in \Lambda_{p-1,q-1} \setminus \{0\}$ for $Q^2 = 0$, is given by

$$
3R^{\frac{q-1}{2}} \bar{G}_{F,\langle\alpha\beta,0}^{(p-1,q-1)\perp}(Q, \varphi) \sum_{l=0}^{1} \frac{\tilde{P}_{\gamma\delta\rangle}^{(l)}(Q)}{R^l} \frac{K_{\frac{q-5}{2}-l}\left(2\pi R\sqrt{2|Q_R|^2}\right)}{\sqrt{2|Q_R|^2}^{\frac{q-3}{2}-l}}
\tag{4.23}
$$

when all the indices are chosen along the sub-Grassmaniann, where $\tilde{P}_{\gamma\delta}^{(l)}$ are defined in (H.1), and where we defined

$$\bar{G}_{F,\alpha\beta,0}^{(p-1,q-1)\perp}(Q,\varphi) = \sum_{\substack{d\geq 1 \\ Q/d\in\Lambda_{p-1,q-1}}} d^{q-6}c(0)\, G_{F,\alpha\beta,0}^{(p-1,q-1)\perp}(\tfrac{Q}{d})\,. \tag{4.24}$$

The full expression for all polarizations will be given together with the polar contributions in (4.44).

Let us point out that $G_{F,\alpha\beta,0}^{(p-1,q-1)\perp}(Q,\varphi) = \frac{\gcd(Q)}{12} G_{\alpha\beta}^{(p-2,q-2)}(\varphi_Q)$ for the function defined in (2.29) for the lattice $\Lambda_{p-2,q-2}$ orthogonal to $Q$, where $\varphi_Q$ parametrizes the Levi subgroup $O(p-2,q-2)$ of the stabilizer of $Q$ in $O(p-1,q-1)$.

2. In the case $\widetilde{Q}_2^2 < 0$, the finite part of the Fourier-Jacobi coefficient has the following expansion in theta series

$$\widehat{\psi}_m^F(\rho,\nu) = \frac{c(m)}{\Delta(\rho)} \sum_{\ell\in\mathbb{Z}_{2m}} \widehat{h}_{m,\ell}(\rho)\theta_{m,\ell}(\rho,\nu)\,, \tag{4.25}$$

where $\theta_{m,\ell}$ and $\widehat{h}_{m,\ell}$ are vector-valued modular forms of weight $1/2$ and $3/2$, respectively defined in (A.62) and (A.67). The integral over $\sigma_1$ enforces $\widetilde{Q}_2^2 = -2m$, while the integral over $u_1$ enforces $\widetilde{Q}_1 \cdot \widetilde{Q}_2 = -\ell$. The summation over $s \in \mathbb{Z}$ in (A.62) can be used to unfold the integral over $u_2 \in [-\frac{1}{2},\frac{1}{2}]$ to the full real axis, after shifting each term in the lattice sum as $\widetilde{Q}_1 \to \widetilde{Q}_1 + s\widetilde{Q}_2$, since $\widetilde{Q}_1,\widetilde{Q}_2 \in \Lambda_{p-1,q-1}$. One thus obtain Fourier coefficients similar to previous case, using $\widetilde{Q}_2 \to Q/k$,

$$3R^{\frac{q-1}{2}} \bar{G}_{F,\langle\alpha\beta,-\frac{Q^2}{2}}^{(p-1,q-1)}(Q,\varphi) \sum_{l=0}^{1} \frac{\tilde{P}_{\gamma\delta\rangle}^{(l)}(Q)}{R^l} \frac{K_{\frac{q-5}{2}-l}\left(2\pi R\sqrt{2|Q_R|^2}\right)}{\sqrt{2|Q_R|^2}^{\frac{q-3}{2}-l}}\,, \tag{4.26}$$

when all the indices are chosen along the sub-Grassmanian, where $\tilde{P}_{\gamma\delta}^{(l)}(Q)$ are defined in (H.1), and where we defined, for $Q^2\neq 0$

$$\bar{G}_{F,\alpha\beta,-\frac{Q^2}{2}}^{(p-1,q-1)}(Q,\varphi) = \sum_{\substack{d\geq 1 \\ Q/d\in\Lambda_{p-1,q-1}}} d^{q-6}c\left(-\tfrac{Q^2}{2d^2}\right)G_{F,\alpha\beta,-\frac{Q^2}{2d^2}}^{(p-1,q-1)\perp}(\tfrac{Q}{d})\,, \tag{4.27}$$

$$G_{F,\alpha\beta,m}^{(p-1,q-1)\perp}(Q) = \int_{\mathcal{F}_1} \frac{d\rho_1 d\rho_2}{\rho_2^2} \frac{1}{\Delta(\rho)} \sum_{\ell\in\mathbb{Z}_{2m}} \widehat{h}_{m,\ell}\Gamma_{\alpha\beta}^{m,\ell}(Q)\,. \tag{4.28}$$

Here $\Gamma_{ab}^{m,\ell}(Q)$ is the lattice partition function (with $\widetilde{Q} = \widetilde{Q}_1 - \frac{\ell}{2m}Q$)

$$\Gamma_{\alpha\beta}^{m,\ell}(Q) = \rho_2^{\frac{q-4}{2}} \sum_{\substack{\widetilde{Q}\in\Lambda_{p-1,q-1}-\frac{\ell}{2m}Q \\ \widetilde{Q}\cdot Q=0}} q^{\frac{1}{2}\widetilde{Q}^2} \phi_{\alpha\beta}^F(\sqrt{2\rho_2}\widetilde{Q},Q)\,, \tag{4.29}$$

with kernel

$$\phi_{\alpha\beta}^F(\sqrt{2\rho_2}\widetilde{Q},Q) = e^{-2\pi\rho_2\left(|\widetilde{Q}_R|^2 - \frac{(\widetilde{Q}_R\cdot Q_R)^2}{|Q_R|^2}\right)}$$
$$\times \left(\rho_2\left(\widetilde{Q}_{L\alpha} - Q_{L\alpha}\frac{\widetilde{Q}_R\cdot Q_R}{|Q_R|^2}\right)\left(\widetilde{Q}_{L\beta} - Q_{L\beta}\frac{\widetilde{Q}_R\cdot Q_R}{|Q_R|^2}\right) - \frac{1}{4\pi}\left(\delta_{\alpha\beta} - \frac{Q_{L\alpha}\cdot Q_{L\beta}}{|Q_R|^2}\right)\right)\,. \tag{4.30}$$

The latter satisfies Vignéras' equation

$$\left(\langle \partial_x, \partial_x \rangle - 2\pi x \partial_x\right)\phi_{\alpha\beta}^F(x, Q) = 2\pi(q-4)\phi_{\alpha\beta}^F(x, Q),\tag{4.31}$$

where $\langle \cdot, \cdot \rangle$ is the inverse of the integer norm on the lattice $\Lambda_{p-1,q-1}$, which ensures [68] that (4.29) is a vector-valued modular form of weight $\frac{p-q+5}{2} = \frac{21}{2}$, consistently with the weight of 3/2 of $\widehat{h}_{m,\ell}(\rho)$ (note that the condition $\widetilde{Q} \cdot Q = 0$ in the sum of (4.29) implies that the lattice over which $\tilde{Q}$ is summed is of dimension $p+q-3$). The analogue expression for other polarizations will be given along with the polar contributions in (4.44).

3. Let us now consider the contributions arising from the polar part $\widehat{\psi}_m^P$ of the Fourier-Jacobi coefficient $\psi_m$ with $m \geq 0$. According to (A.69), the latter can be written as an indefinite theta series

$$\widehat{\psi}_m^P(\rho, \nu) = \frac{c(m)}{\Delta(\rho)} \sum_{s,\ell \in \mathbb{Z}} q^{ms^2 + s\ell} y^{2ms+\ell} \widehat{b}(s, \ell, m, \rho_2),\tag{4.32}$$

where for $m \geq 1$,

$$\widehat{b}(s, \ell, m, \rho_2) = \frac{1}{2}\ell \left[ \mathrm{sgn}(s + u_2) + \mathrm{erf}\left(\ell\sqrt{\frac{\pi\rho_2}{m}}\right) \right] + \frac{\sqrt{m}}{2\pi\sqrt{\rho_2}} e^{-\pi\rho_2\ell^2/m} - \frac{1}{4\pi\rho_2}\delta(s + u_2),\tag{4.33}$$

whereas

$$\widehat{b}(s, \ell, 0, \rho_2) = \frac{1}{2}\ell \left[ \mathrm{sgn}(s + u_2) + \mathrm{sgn}(\ell) \right] - \frac{1}{4\pi\rho_2}\delta(s + u_2) + \delta_{s,0}\delta_{\ell,0}\frac{1}{4\pi\rho_2}.\tag{4.34}$$

As in the previous case, one can shift the charges to $\widetilde{Q}_1 \to \widetilde{Q}_1 + s\widetilde{Q}_2$ since $\widetilde{Q}_1, \widetilde{Q}_2 \in \Lambda_{p-1,q-1}$, and then use the sum over $s$ to unfold the $u_2 \in [-\frac{1}{2}, \frac{1}{2}]$ to $\mathbb{R}$. Then, integrating over $u_1 \in [-\frac{1}{2}, \frac{1}{2}]$ imposes $\widetilde{Q}_1 \cdot \widetilde{Q}_2 = -\ell$. One then carries out the change of variable $u_2 = \frac{u}{\sqrt{2\rho_2|Q_R|^2}}$. One obtains the Fourier coefficients, using $\widetilde{Q}_2 \to Q/k$,

$$3R^{\frac{q-1}{2}} \bar{G}_{P,\langle\alpha\beta,}^{(p-1,q-1)}(Q) \sum_{l=0}^{1} \frac{\tilde{P}_{\gamma\delta\rangle}^{(l)}(Q)}{R^l} \frac{K_{\frac{q-5}{2}-l}\left(2\pi R\sqrt{2|Q_R|^2}\right)}{\sqrt{2|Q_R|^2}^{\frac{q-3}{2}-l}},\tag{4.35}$$

when all indices are chosen along the sub-Grassmanian, and where we define for $Q^2 < 0$

$$\bar{G}_{P,\alpha\beta}^{(p-1,q-1)}(Q, \varphi) = \sum_{\substack{d \geq 1 \\ Q/d \in \Lambda_{p-1,q-1}}} d^{q-6} c(-\tfrac{Q^2}{2d^2}) G_{P,\alpha\beta}^{(p-1,q-1)}(\tfrac{Q}{d}),\tag{4.36}$$

$$G_{P,\alpha\beta}^{(p-1,q-1)}(Q) = \int_{\mathcal{F}_1} \frac{d\rho_1 d\rho_2}{\rho_2^2} \frac{\rho_2^{\frac{q-5}{2}}}{\Delta(\rho)} \sum_{\widetilde{Q} \in \Lambda_{p-1,q-1}} q^{\frac{1}{2}\widetilde{Q}^2} \phi_{P,\alpha\beta}(\sqrt{2\rho_2}\widetilde{Q}, Q),\tag{4.37}$$

with the kernel

$$\phi_{P,\alpha\beta}(x, Q) = -\frac{1}{4\sqrt{2}} \int_{\mathbb{R}} du\,(x \cdot Q)\left[ \mathrm{sgn}(u) + \mathrm{erf}\left(-\sqrt{\frac{\pi}{-Q^2}}\,x \cdot Q\right) - \frac{\sqrt{-Q^2}}{\pi x \cdot Q} e^{\frac{\pi(x \cdot Q)^2}{Q^2}} \right]$$
$$\times e^{-\pi|x_R|^2 - \pi u^2 - 2\pi u \frac{x_R \cdot Q_R}{|Q_R|}} \left( \left(x_{L\alpha} + u\frac{Q_{L\alpha}}{|Q_R|}\right)\left(x_{L\beta} + u\frac{Q_{L\beta}}{|Q_R|}\right) - \frac{1}{2\pi}\delta_{\alpha\beta} \right)$$
$$- \frac{\sqrt{2|Q_R|^2}}{8\pi} e^{-\pi|x_R|^2} \left( x_{L\alpha} x_{L\beta} - \frac{1}{2\pi}\delta_{\alpha\beta} \right).\tag{4.38}$$

Using integration by part over $u$ one computes that $\phi_{P,\alpha\beta}(x,Q)$ satisfies the Vignéras equation

$$\left(\langle\partial_x,\partial_x\rangle-2\pi x\partial_x\right)\phi_{P,\alpha\beta}(x,Q)=2\pi(q-5)\phi_{P,\alpha\beta}(x,Q)\,,\tag{4.39}$$

therefore the lattice sum in (4.37) is a modular form of weight $\frac{p-q}{2}+4$, and the integral is well defined. For $Q^2=0$, one has instead

$$G_{P,\alpha\beta}^{(p-1,q-1)}(Q)=\int_{\mathcal{F}_1}\frac{\mathrm{d}\rho_1\mathrm{d}\rho_2}{\rho_2^2}\frac{\rho_2^{\frac{q-5}{2}}}{\Delta(\rho)}\Bigg(\sum_{\widetilde{Q}\in\Lambda_{p-1,q-1}}q^{\frac{1}{2}\widetilde{Q}^2}\phi_{P,\alpha\beta}(\sqrt{2\rho_2}\widetilde{Q},Q)$$
$$+\rho_2^{-\frac{1}{2}}\sum_{\widetilde{Q}\in\Lambda_{p-2,q-2}}q^{\frac{1}{2}\widetilde{Q}^2}\phi_{P,\alpha\beta}'^{\perp}(\sqrt{2\rho_2}\widetilde{Q},Q)\Bigg)\,,\tag{4.40}$$

with

$$\phi_{P,\alpha\beta}(x,Q)=-\frac{1}{4\sqrt{2}}\int_{\mathbb{R}}\mathrm{d}u\left((x\cdot Q)\mathrm{sgn}(u)-|x\cdot Q|\right)e^{-\pi|x_R|^2-\pi u^2-2\pi u\frac{x_R\cdot Q_R}{|Q_R|}}$$
$$\times\left(\left(x_{L\alpha}+u\frac{Q_{L\alpha}}{|Q_R|}\right)\left(x_{L\beta}+u\frac{Q_{L\beta}}{|Q_R|}\right)-\frac{1}{2\pi}\delta_{\alpha\beta}\right)$$
$$-\frac{\sqrt{2|Q_R|^2}}{8\pi}e^{-\pi|x_R|^2}\left(x_{L\alpha}x_{L\beta}-\frac{1}{2\pi}\delta_{\alpha\beta}\right)\,,\tag{4.41}$$

$$\phi_{P,\alpha\beta}'^{\perp}(x,Q)=\frac{e^{-\pi(x_R^2-\frac{(x_R\cdot Q_R)^2}{Q_R^2})}}{8\pi}\left(\left(x_{L\alpha}-\frac{x_R\cdot Q_R}{Q_R^2}Q_{L\alpha}\right)\left(x_{L\beta}-\frac{x_R\cdot Q_R}{Q_R^2}Q_{L\beta}\right)-\frac{1}{2\pi}\left(\delta_{\alpha\beta}-\frac{Q_{L\alpha}Q_{L\beta}}{Q_R^2}\right)\right)\,.$$

The integrand in (4.40) must be modular by construction, but its modularity does not follow directly from Vignéras theorem. In this case $\phi_{P,\alpha\beta}(x,Q)$ satisfies Vignéras equation (4.39), but it is a distribution and its second derivative is not square integrable. The function $\phi_{P,\alpha\beta}'^{\perp}(x,Q)$ satisfies Vignéras equation (4.31), but this is not the correct eigenvalue to give the correct modular weight. As the failure of $\phi_{P,\alpha\beta}(x,Q)$ to define a modular form comes from its singularity at $(Q\cdot x)=0$, it is somehow natural that its modular anomaly can be compensated by a partition function on the lattice orthogonal to $Q$.

4. Finally, the case $m=-1$ requires special treatment. The finite part of $\psi_{-1}$ automatically vanishes, but the polar part is proportional to a modified Appell-Lerch sum, as explained in Appendix A.5,

$$\psi_{-1}=-\frac{1}{\Delta}\sum_{s,\ell\in\mathbb{Z}}\left[\ell\frac{\mathrm{sign}(\ell-2s)+\mathrm{sign}(u_2+s)}{2}-\frac{1}{4\pi\rho_2}\delta(u_2+s)\right]q^{-s^2+\ell s}y^{\ell-2s}\,,\tag{4.42}$$

which differs from the naive Appell-Lerch sum (which diverges when the index is negative) by a replacement $\mathrm{sign}\ell\to\mathrm{sign}(\ell-2s)$. In this case we still get (4.40) with

$$\phi_{P,\alpha\beta}(x,Q)=-\frac{1}{4\sqrt{2}}\int_{\mathbb{R}}\mathrm{d}u\,(x\cdot Q)\left[\mathrm{sgn}(u)-\mathrm{sign}\left(\frac{x\cdot Q}{\sqrt{2\rho_2}}+2\lfloor\frac{u}{\sqrt{2\rho_2 Q_R^2}}\rfloor\right)\right]$$
$$\times e^{-\pi|x_R|^2-\pi u^2-2\pi u\frac{x_R\cdot Q_R}{|Q_R|}}\left(\left(x_{L\alpha}+u\frac{Q_{L\alpha}}{|Q_R|}\right)\left(x_{L\beta}+u\frac{Q_{L\beta}}{|Q_R|}\right)-\frac{1}{2\pi}\delta_{\alpha\beta}\right)$$
$$-\frac{\sqrt{2|Q_R|^2}}{8\pi}e^{-\pi|x_R|^2}\left(x_{L\alpha}x_{L\beta}-\frac{1}{2\pi}\delta_{\alpha\beta}\right)\,,\tag{4.43}$$

$$\phi_{P,\alpha\beta}'^{\perp}(x,Q)=\frac{e^{-\pi(x_R^2-\frac{(x_R\cdot Q_R)^2}{Q_R^2})}}{8\pi}\left(\left(x_{L\alpha}-\frac{x_R\cdot Q_R}{Q_R^2}Q_{L\alpha}\right)\left(x_{L\beta}-\frac{x_R\cdot Q_R}{Q_R^2}Q_{L\beta}\right)-\frac{1}{2\pi}\left(\delta_{\alpha\beta}-\frac{Q_{L\alpha}Q_{L\beta}}{Q_R^2}\right)\right)\,.$$

Although the modularity of (4.43) no longer follows from Vignéras' theorem, it must hold by construction.

Combining the finite and polar contributions, we finally obtain the full expressions for the exponentially suppressed corrections,

$$
G_{\alpha\beta,\gamma\delta}^{(p,q),1,Q} = 3R^{\frac{q-1}{2}} \bar{G}_{\langle\alpha\beta,}^{(p-1,q-1)}(Q,\varphi) \sum_{l=0}^{1} \frac{\tilde{P}_{\gamma\delta\rangle}^{(l)}(Q)}{R^l} \frac{K_{\frac{q-5}{2}-l}\left(2\pi R\sqrt{2Q_R^2}\right)}{(2Q_R^2)^{\frac{q-3-2l}{4}}} ,
$$

$$
G_{\alpha\beta,\gamma 1}^{(p,q),1,Q} = \frac{3}{2} R^{\frac{q-1}{2}} \bar{G}_{\langle\alpha\beta,}^{(p-1,q-1)}(Q,\varphi) \frac{Q_{L\gamma\rangle}}{\mathrm{i}\sqrt{2}} \frac{K_{\frac{q-7}{2}}\left(2\pi R\sqrt{2Q_R^2}\right)}{(2Q_R^2)^{\frac{q-5}{4}}} , \tag{4.44}
$$

$$
G_{\alpha\beta,11}^{(p,q),1,Q} = -R^{\frac{q-1}{2}} \bar{G}_{\alpha\beta}^{(p-1,q-1)}(Q,\varphi) \frac{K_{\frac{q-9}{2}}\left(2\pi R\sqrt{2Q_R^2}\right)}{(2Q_R^2)^{\frac{q-7}{4}}} ,
$$

where the polynomials $\tilde{P}_{\gamma\delta}^{(l)}(Q)$ are given in (H.1), and the coefficient $\bar{G}_{\alpha\beta}^{(p-1,q-1)}$ is defined by

$$
\bar{G}_{\alpha\beta}^{(p-1,q-1)}(Q,\varphi) = \sum_{\substack{d\geq 1 \\ Q/d\in\Lambda_{p-1,q-1}}} d^{q-6} c(-\tfrac{Q^2}{2d^2})\Big( G_{F,\alpha\beta,-\frac{Q^2}{2d^2}}^{(p-1,q-1)\perp}(\tfrac{Q}{d}) + G_{P,\alpha\beta}^{(p-1,q-1)}(\tfrac{Q}{d})\Big), \tag{4.45}
$$

where $G_F^{(p-1,q-1)}$ and $G_P^{(p-1,q-1)}$ are defined in (4.24), (4.28),(4.36) for $Q^2 < 0$, in (4.40) for $Q^2 = 0$ and in (4.43) for $Q^2 > 0$.

## 4.2 Extension to $\mathbb{Z}_N$ CHL orbifolds

The degeneration limit (4.2) of the modular integral (2.30) for $\mathbb{Z}_N$ CHL models with $N = 2, 3, 5, 7$ can be treated similarly by adapting the orbit method to the case where the integrand is invariant under the congruence subgroup $\Gamma_{2,0}(N) = \{\left(\begin{smallmatrix} A & B \\ C & D \end{smallmatrix}\right) \in Sp(4,\mathbb{Z}), C = 0 \bmod N\}$. In (2.30), $\Phi_{k-2}$ is the meromorphic Siegel modular form of $\Gamma_{2,0}(N)$ of weight $k-2$ defined in §A.4, and $\Gamma_{\Lambda_{p,q}}^{(2)}$ is the genus-two partition function for a lattice

$$
\Lambda_{p,q} = \Lambda_{p-1,q-1} \oplus I\!I_{1,1}[N] , \tag{4.46}
$$

where $\Lambda_{p-1,q-1}$ is a level $N$ even lattice of signature $(p-1,q-1)$. The lattice $I\!I_{1,1}[N]$ is obtained from the usual unimodular lattice $I\!I_{1,1}$ by restricting the winding and momentum to $(n_1, n_2, m_1, m_2) \in N\mathbb{Z} \oplus N\mathbb{Z} \oplus \mathbb{Z} \oplus \mathbb{Z}$. After Poisson resummation on $m_1, m_2$, Eq. (4.7) continues to hold, except for the fact that $n_1, n_2$ are restricted to run over $N\mathbb{Z}$. The sum over $(n_1, n_2, m_1, m_2)$ can then be decomposed into orbits of $\Gamma_{2,0}(N)$:

**Trivial orbit** The term $(n_1, n_2, m_1, m_2) = (0,0,0,0)$ produces the same modular integral, up to a factor of $R^2$,

$$
G_{\alpha\beta,\gamma\delta}^{(p,q),0} = R^2 G_{\alpha\beta,\gamma\delta}^{(p-1,q-1)} , \tag{4.47}
$$

where $G_{\alpha\beta,\gamma\delta}^{(p-1,q-1)}$, is the integral (2.30) for the lattice $\Lambda_{p-1,q-1}$ defined by (4.46).

**Rank-one orbits** Terms with $(n_1, n_2, m_1, m_2) = k(c_3, c_4, d_3, d_4)$ with $k \neq 0$ and $\gcd(c_3, c_4, d_3, d_4) = 1$ fall into two different classes of orbits under $\Gamma_{2,0}(N)$:

1. Quadruplets $k(c_3, c_4, d_3, d_4)$ such that $(c_3, c_4) = (0,0) \bmod N$ and $k \in \mathbb{Z}$ can be rotated by an element of $\Gamma_{2,0}(N)$ into $(0,0,0,1)$, whose stabilizer in $\Gamma_{2,0}(N)$ is $\Gamma_0(N) \ltimes H_{2,1}(\mathbb{Z}) \subset \Gamma_1^J$. For these elements, one can unfold the integration domain $\Gamma_{2,0}(N) \backslash \mathcal{H}_2$ into the domain

$$(\Gamma_0(N) \ltimes H_{2,1}(\mathbb{Z})) \backslash \mathcal{H}_2 = \mathbb{R}_t^+ \times (\Gamma_0(N) \backslash \mathcal{H}_1)_\rho \times \left((\mathbb{R}/\mathbb{Z})^3/\mathbb{Z}_2\right)_{u_1, u_2, \sigma_1}, \tag{4.48}$$

where the $\mathbb{Z}_2$ comes from $-\mathbb{1} \in \Gamma_0(N)$ leaving $\rho$ invariant but acting as $(u_1, u_2) \rightarrow (-u_1, u_2)$ on the other moduli.

2. Doublets $k(c_3, c_4, d_3, d_4)$ such that $(c_3, c_4) \neq (0,0) \bmod N$ have $k = 0 \bmod N$ since $(n_1, n_2) = 0 \bmod N$. They can be rotated by an element of $\Gamma_{2,0}(N)$ into $(0,1,0,0)$, whose stabilizer in $\Gamma_{2,0}(N)$ is $S_\rho S_\sigma \left(\Gamma^0(N) \ltimes H_{2,1,N}^{(2)}(\mathbb{Z})\right)(S_\rho S_\sigma)^{-1}$, where

$$H_{2,1,N}^{(2)}(\mathbb{Z}) = \{(\kappa, \lambda, \mu) \in H_{2,1}(\mathbb{Z}), \, \kappa = \mu = 0 \bmod N\}, \tag{4.49}$$

and the inversion on $\sigma$ is $S_\sigma : (\rho, \sigma, \nu) \rightarrow (\rho - \nu^2/\sigma, -1/\sigma, -\nu/\sigma)$. One can unfold the integration domain $\Gamma_{2,0}(N) \backslash \mathcal{H}_2$ into $S_\rho S_\sigma \left(\Gamma^0(N) \ltimes H_{2,1,N}^{(2)}(\mathbb{Z})\right)(S_\rho S_\sigma)^{-1} \backslash \mathcal{H}_2$, and change variable

$$\Omega \rightarrow (S_\rho S_\sigma) \cdot \Omega = -\Omega^{-1}, \tag{4.50}$$

so as to reach $(\Gamma^0(N) \ltimes H_{2,1,N}^{(2)}(\mathbb{Z})) \backslash \mathcal{H}_2 = \frac{1}{2}\mathbb{R}_t^+ \times (\Gamma^0(N) \backslash \mathcal{H}_1)_\rho \times (\mathbb{R}/\mathbb{Z})_{u_2} \times (\mathbb{R}/N\mathbb{Z})_{u_1, \sigma_1}^2$. Under this change of variable, the level-$N$ weight-$(k-2)$ Siegel modular form transforms as

$$\Phi_{k-2}(-\Omega^{-1}) = (i\sqrt{N})^{-2(k-2)}|\Omega|^{k-2}\Phi_{k-2}(\Omega/N), \tag{4.51}$$

while the genus-two partition function for the sublattice $\Lambda_{p-1,q-1}$ transforms as

$$\Gamma_{\tilde{\Lambda}_{p-1,q-1}}^{(2)}[P_{\alpha\beta,\gamma\delta}](-\Omega^{-1}) = v^2 N^{-k-2}(-i)^{p-q}|\Omega|^{k-2}\Gamma_{\tilde{\Lambda}_{p-1,q-1}^*}^{(2)}[P_{\alpha\beta,\gamma\delta}](\Omega), \tag{4.52}$$

where we denoted $v^2 N^{-k-2} = \left|\Lambda_{p-1,q-1}^*/\Lambda_{p-1,q-1}\right|^{-1}$ the volume factor from Poisson ressummation (Note that $v^2 = N^{2-2\delta_{q,8}}$ for $q \leq 8$ in the cases of interest).

For the function $G_{ab,cd}^{(p,q),1}$, changing $y$ variables as before $(y'_{11}, y'_{21}, y'_{1\alpha}, y'_{2\alpha}) = (y_{11}, y_{11}u_2 - y_{21}, y_{1\alpha}, y_{1\alpha}u_2 - y_{2\alpha})$, the sum of the two classes of orbits then reads

$$
\begin{aligned}
G_{ab,cd}^{(p,q),1} = \frac{R^2}{2} \int_{\mathbb{R}^+} \frac{dt}{t^3} \int_{(\mathbb{R}/\mathbb{Z})^3} du_1 du_2 d\sigma_1 \int_{\Gamma_0(N)\backslash\mathcal{H}_1} \frac{d\rho_1 d\rho_2}{\rho_2^2} \frac{\mathcal{P}_{ab,cd}(\frac{\partial}{\partial y'})}{\Phi_{k-2}(\Omega)} \\
\times \sum_{k\neq 0} e^{-\frac{\pi R^2 k^2}{t}} \Gamma_{\tilde{\Lambda}_{p-1,q-1}}^{(2)}\left[e^{2\pi i k a^I \tilde{Q}_{2I}} \mathcal{Y}(y')\right] \\
+ \frac{R^2}{2} \int_{\mathbb{R}^+} \frac{dt}{t^3} \int_{(\mathbb{R}/N\mathbb{Z})^2} du_1 d\sigma_1 \int_{\mathbb{R}/\mathbb{Z}} du_2 \int_{\Gamma^0(N)\backslash\mathcal{H}_1} \frac{d\rho_1 d\rho_2}{\rho_2^2} \frac{\mathcal{P}_{ab,cd}(\frac{\partial}{\partial y'})}{\Phi_{k-2}(\Omega/N)} \\
\times \frac{v^2}{N^4} \sum_{\substack{k\neq 0 \\ k=0 \bmod N}} e^{-\frac{\pi R^2 k^2}{t}} \Gamma_{\tilde{\Lambda}_{p-1,q-1}^*}^{(2)}\left[e^{2\pi i k a^I \tilde{Q}_{2I}} \mathcal{Y}(y')\right],
\end{aligned}
\tag{4.53}
$$

where

$$\mathcal{Y}(y') = e^{2\pi i\left(\frac{R}{i\sqrt{2}}\frac{k}{t}y'_{21} + y'_{1\alpha}(Q_L^{1\alpha} + u_2 Q_L^{2\alpha}) - y'_{2\alpha}Q_L^{2\alpha} + \frac{1}{4i\rho_2}y'_{1\alpha}y_1'^{\alpha} + \frac{1}{4it}y'_{2\alpha}y_2'^{\alpha}\right)}. \tag{4.54}$$

As before, we substitute $1/\Phi_{k-2}$ by its Fourier-Jacobi expansion $1/\Phi_{k-2} = \sum_{m \geq -1} \psi_{k-2,m} e^{2\pi i m \sigma}$, so that the integral over $\sigma_1$ enforces $\widetilde{Q}_2^2 = -2m$. For $\widetilde{Q}_2^2 = 0$ case, the integral over $u_1, u_2$ in the first line follows from (A.73),

$$\int_{-\frac{1}{2}}^{\frac{1}{2}} du_1 \int_{-\frac{1}{2}}^{\frac{1}{2}} du_2 \, \psi_{k-2,0}(\rho, u_1 + \rho u_2) = \frac{c_k(0)}{12(N-1)} \frac{N^2 \widehat{E}_2(N\rho) - \widehat{E}_2(\rho)}{\Delta_k(\rho)}, \quad (4.55)$$

where $N^2 \widehat{E}_2(N\rho) - \widehat{E}_2(\rho)$ is a level-$N$ weight 2 holomorphic modular form. The contribution from the second line in (4.53) is calculated using the transformation properties of the genus-one cusp form and partition function[11]. The transformation $\rho \to -1/\rho$ changes the integration domain from $\Gamma^0(N) \backslash \mathcal{H}_1$ to $\Gamma_0(N) \backslash \mathcal{H}_1$, and one thus obtains, denoting $Q_I = k \widetilde{Q}_{2I}$

$$G_{ab,cd}^{(p,q),1,Q^2=0} = R^2 \int_0^\infty \frac{dt}{t} t^{\frac{q-5}{2}} \sum_{\substack{\widetilde{Q}_2 \in \Lambda_{p-1,q-1} \\ \widetilde{Q}_2^2 = 0}} \sum_{k \neq 0} e^{-\frac{\pi R^2 k^2}{t} - 2\pi t Q_R^2/k^2} \frac{c_k(0)}{24(N-1)}$$

$$\times \int_{\Gamma_0(N) \backslash \mathcal{H}_1} \frac{d\rho_1 d\rho_2}{\rho_2^2} \frac{(N^2 - \upsilon N^{q-6}) \widehat{E}_2(N\rho) + (\upsilon N^{q-6} - 1) \widehat{E}_2(\rho)}{\Delta_k(\rho)} \Gamma_{\widetilde{\Lambda}_{p-1,q-1}} \Big[ e^{2\pi i a^I Q_I} \mathcal{P}_{ab,cd}(\tfrac{\partial}{\partial y'}) \mathcal{Y}(y') \Big]. \quad (4.56)$$

The zero mode contribution, $Q = 0$, may be expressed in terms of the genus-one modular integrals

$$G_{ab}^{(p,q)} = \text{R.N.} \int_{\Gamma_0(N) \backslash \mathcal{H}_1} \frac{d\rho_1 d\rho_2}{\rho_2^2} \frac{\widehat{E}_2 \, \Gamma_{\Lambda_{p,q}}[P_{ab}]}{\Delta_k}, \quad (4.57)$$

$$\,^\varsigma G_{ab}^{(p,q)} = \text{R.N.} \int_{\Gamma_0(N) \backslash \mathcal{H}_1} \frac{d\rho_1 d\rho_2}{\rho_2^2} \frac{N \widehat{E}_2(N\rho)}{\Delta_k(\rho)} \Gamma_{\Lambda_{p,q}}[P_{ab}]. \quad (4.58)$$

When $\Lambda_{p,q}$ is $N$-modular, such that $\Lambda_{p,q}^* = \varsigma \cdot \Lambda_{p,q}/\sqrt{N}$ for $\varsigma \in O(p,q,\mathbb{R})$, then $\,^\varsigma G_{ab}^{(p,q)} = G_{ab}^{(p,q)}(\varsigma \cdot \varphi)$. The zero mode $Q = 0$ thus leads to power-like terms

$$G_{\alpha\beta,\gamma\delta}^{(p,q),1,0} = -R^{q-5} \xi(q-6) \frac{c_k(0)}{16\pi} \Big[ \frac{\upsilon N^{q-6} - 1}{N-1} \delta_{\langle \alpha\beta,} G_{\gamma\delta \rangle}^{(p-1,q-1)} + \frac{N - \upsilon N^{q-7}}{N-1} \delta_{\langle \alpha\beta,} \,^\varsigma G_{\gamma\delta \rangle}^{(p-1,q-1)} \Big],$$

$$G_{\alpha\beta,11}^{(p,q),1,0} = -R^{q-5} \xi(q-6)(7-q) \frac{c_k(0)}{48\pi} \Big[ \frac{\upsilon N^{q-6} - 1}{N-1} G_{\alpha\beta}^{(p-1,q-1)} + \frac{N - \upsilon N^{q-7}}{N-1} \,^\varsigma G_{\alpha\beta}^{(p-1,q-1)} \Big]. \quad (4.59)$$

As in the maximal rank case (4.20), the unfolding trick fails to capture another powerlike term proportional to $R^{2q-12}$, which is required by the non-homogeneous differential equation (3.20). This term can be seen to arise in the maximal non-separating degeneration, and can be computed as in (4.19), leading to

$$G_{\alpha\beta,\gamma\delta}^{(p,q),1,0'} = -\frac{3}{64\pi^3} R^{2q-12} \big[ c_k(0)(1 + \upsilon N^{q-7})\xi(q-6) \big]^2 \delta_{\langle \alpha\beta,} \delta_{\gamma\delta \rangle},$$

$$G_{\alpha\beta,11}^{(p,q),1,0'} = -\frac{1}{32\pi^3} R^{2q-12} \big[ c_k(0)(1 + \upsilon N^{q-7})\xi(q-6) \big]^2 (7-q)\delta_{\alpha\beta}. \quad (4.60)$$

These results can also be obtained by taking the limit $S_2 \to \infty$ from the result (5.60) obtained in the degeneration limit $(p,q) \to (p-2, q-2)$.

---

[11]*i.e.* $\Delta_k(-1/N\rho) = N^{\frac{k}{2}}(-i\rho)^k \Delta_k(\rho)$, and $\Gamma_{\Lambda_{p-1,q-1}^*}[P_{ab}](-1/\rho) = \upsilon^{-1} N^{\frac{k}{2}+1}(-i)^k \rho^{k-2} \Gamma_{\Lambda_{p-1,q-1}}[P_{ab}](\rho)$ where $\upsilon = N^{\frac{k}{2}+1} \big| \Lambda_{p-1,q-1}^* / \Lambda_{p-1,q-1} \big|^{-1/2}$

The contributions from vectors $Q \neq 0$ lead to exponentially suppressed contributions of the same form as the Fourier modes of null vectors (4.23), non-null vectors (4.26), and the polar contribution (4.36) respectively, with different coefficients:

1. For null Fourier vectors $Q^2 = 0$, the moduli-dependent coefficient coming from the finite part of $1/\Phi_{k-2}(\Omega)$ reads

$$
\bar{G}_{F,\alpha\beta,0}^{(p-1,q-1)}(Q,\varphi) = \sum_{\substack{d>0 \\ Q/d\in\Lambda_{p-1,q-1}}} d^{q-6} c_k(0) \frac{N \,{}^{\varsigma}G_{F,\alpha\beta,0}^{(p-1,q-1)\perp}\!\left(\frac{Q}{d}\right) - G_{F,\alpha\beta,0}^{(p-1,q-1)\perp}\!\left(\frac{Q}{d}\right)}{N-1}
$$
$$
+ \upsilon \sum_{\substack{d>0 \\ Q/d\in N\Lambda_{p-1,q-1}^*}} (Nd)^{q-6} c_k(0) \frac{N G_{F,\alpha\beta,0}^{(p-1,q-1)\perp}\!\left(\frac{Q}{Nd}\right) - {}^{\varsigma}G_{F,\alpha\beta,0}^{(p-1,q-1)\perp}\!\left(\frac{Q}{Nd}\right)}{N-1}, \quad (4.61)
$$

where $G_{F,ab,0}^{(p,q)}(\varphi)$ is defined as in (4.22) with $\hat{E}_2/\Delta$ replaced by $\hat{E}_2/\Delta_k$, and ${}^{\varsigma}G_{F,ab,0}^{(p,q)}(\varphi)$ is defined as in (4.22) with $\hat{E}_2/\Delta$ replaced by $N\hat{E}_2(N\rho)/\Delta_k(\rho)$.

2. For non-null Fourier vectors, $Q^2 \neq 0$, the moduli-dependent coefficient coming from the finite part of $1/\Phi_{k-2}(\Omega)$ is given by

$$
\bar{G}_{F,\alpha\beta,-\frac{Q^2}{2}}^{(p-1,q-1)}(Q,\varphi) = \sum_{\substack{d>0 \\ Q/d\in\Lambda_{p-1,q-1}}} d^{q-6} c_k\!\left(-\frac{Q^2}{2d^2}\right) G_{F,\alpha\beta,-\frac{Q^2}{2d^2}}^{(p-1,q-1)\perp}\!\left(\frac{Q}{d}\right)
$$
$$
+ \upsilon \sum_{\substack{d>0 \\ Q/d\in N\Lambda_{p-1,q-1}^*}} (Nd)^{q-6} c_k\!\left(-\frac{Q^2}{2Nd^2}\right) {}^{\varsigma}G_{F,\alpha\beta,-\frac{Q^2}{2Nd^2}}^{(p-1,q-1)\perp}\!\left(\frac{Q}{Nd}\right), \quad (4.62)
$$

where we defined, similarly to ${}^{\varsigma}G_{F,\alpha\beta,0}^{(p-1,q-1)}(Q)$,

$$
{}^{\varsigma}G_{F,\alpha\beta,m}^{(p-1,q-1)}(Q) = \int_{\Gamma_0(N)\backslash\mathcal{H}_1} \frac{d^2\rho}{\rho_2^2} \sum_{l\in\mathbb{Z}_{2m}} \frac{N\widehat{h}_{m,l}(N\rho)}{\Delta_k(\rho)} \Gamma_{\alpha\beta}^{m,l}(Q), \quad (4.63)
$$

with $\Gamma_{\alpha\beta}^{m,l}(Q)$ defined in (4.29).

3. For all non-zero vectors $Q \neq 0$, the moduli-dependent coefficient coming from the polar part of $1/\Phi_{k-2}(\Omega)$ is given by

$$
\bar{G}_{P,\alpha\beta}^{(p-1,q-1)}(Q,\varphi) = \sum_{\substack{d>0 \\ Q/d\in\Lambda_{p-1,q-1}}} d^{q-6} c_k\!\left(-\frac{Q^2}{2d^2}\right) G_{P,\alpha\beta}^{(p-1,q-1)}\!\left(\frac{Q}{d}\right)
$$
$$
+ \upsilon \sum_{\substack{d>0 \\ Q/d\in N\Lambda_{p-1,q-1}^*}} (Nd)^{q-6} c_k\!\left(-\frac{Q^2}{2Nd^2}\right) G_{P,\alpha\beta}^{(p-1,q-1)}\!\left(\frac{Q}{Nd}\right), \quad (4.64)
$$

where $G_{P,\alpha\beta}^{(p-1,q-1)}$ is defined as in the previous subsection, upon replacing $\Delta(\rho)$ by $\Delta_k(\rho)$.

Note that the polar part and the finite part of the function $\bar{G}_{\alpha\beta}^{(p-1,q-1)}(Q,\varphi)$ combine for all $Q$ into the same divisor sum of the function $G_{\alpha\beta}^{(p-1,q-1)}(Q) = G_{F\alpha\beta}^{(p-1,q-1)}(Q) + G_{P\alpha\beta}^{(p-1,q-1)}(Q)$ and ${}^{\varsigma}G_{\alpha\beta}^{(p-1,q-1)}(Q) = {}^{\varsigma}G_{F\alpha\beta}^{(p-1,q-1)}(Q) + {}^{\varsigma}G_{P\alpha\beta}^{(p-1,q-1)}(Q)$ as in the maximal rank case (4.45). The only apparent difference is for the finite part of the function (4.61), because we defined the function (4.61) $G_{F,\alpha\beta,0}^{(p-1,q-1)\perp}\!\left(\frac{Q}{d}\right)$ and ${}^{\varsigma}G_{F,\alpha\beta,0}^{(p-1,q-1)\perp}\!\left(\frac{Q}{d}\right)$ such that they can be identified to the function $\frac{\gcd(Q)}{12} G_{\alpha\beta}^{(p-2,q-2)}(\varphi_Q)$ and $\frac{\gcd(Q)}{12} {}^{\varsigma}G_{\alpha\beta}^{(p-2,q-2)}(\varphi_Q)$ on the quotient of the sublattice of $\Lambda_{p-1,q-1}$ orthogonal to $Q$ by the shift in $Q$.

## 4.3 Perturbative limit of exact heterotic $\nabla^2(\nabla\phi)^4$ couplings in $D=3$

According to our Ansatz (1.7), the exact $\nabla^2(\nabla\phi)^4$ coupling in three-dimensional CHL orbifolds is given by a special case of the family of genus-two modular integrals (4.1) for the 'non-perturbative Narain lattice' (2.3) of signature $(p,q) = (2k,8) = (2k,8)$. The degeneration (4.2) studied in this section corresponds to the limit of weak heterotic coupling $g_3 \to 0$. In this limit, the lattice $\Lambda_{2k,8}$ decomposes into $\Lambda_{2k-1,7} \oplus I\!I_{1,1}[N]$, where the 'radius' of the second factor is related to the heterotic string coupling by $g_3 = 1/\sqrt{R}$, and the U-duality group is broken to $\widetilde{O}(2k-1,7,\mathbb{Z}) \subset \widetilde{O}(2k,8,\mathbb{Z})$, with $\widetilde{O}(2k-1,7,\mathbb{Z})$ the restricted automorphic group of $\Lambda_{2k-1,7} = \Lambda_m \oplus I\!I_{1,1}[N]$. In order to interpret the various power-like terms in the large radius expansion as perturbative contributions to the $\nabla^2(\nabla\phi)^4$ coupling, it is convenient to multiply the coupling by a factor of $g_3^6$, which arises due to the Weyl rescaling $\gamma_E = \gamma_s/g_3^4$ from the Einstein frame to the string frame [22, Sec 4.3]. The weak coupling expansion can be extracted from section 4.2 upon setting $q=8$ and $\upsilon=1$, and reads

$$
\begin{aligned}
g_3^6 G_{\alpha\beta,\gamma\delta}^{(2k,8)} =\ & -\frac{3}{4\pi g_3^2}\delta_{\langle\alpha\beta},\delta_{\gamma\delta\rangle} - \frac{1}{4}\delta_{\langle\alpha\beta},G_{\gamma\delta\rangle}^{(2k-1,7)}(\varphi) + g_3^2\, G_{\alpha\beta,\gamma\delta}^{(2k-1,7)}(\varphi) \\
& + \sum_{Q\in\Lambda_{2k-1,7}^*}^{\prime} \frac{3e^{-\frac{2\pi}{g_3^2}\sqrt{2Q_R^2}+2\pi iQ\cdot a}}{2Q_R^2}\, \bar{G}_{\langle\alpha\beta,}^{(2k-1,7)}(Q,\varphi)\Big(Q_{L\gamma}Q_{L\delta\rangle}\Big(\sqrt{2Q_R^2}+\frac{g_3^2}{2\pi}\Big)-\frac{g_3^2}{8\pi}\delta_{\gamma\delta\rangle}\Big) \\
& + \sum_{Q\in\Lambda_{2k-1,7}^*}^{\prime} e^{-\frac{4\pi}{g_3^2}\sqrt{2Q_R^2}}G_{\alpha\beta,\gamma\delta}(g_3,Q_L,Q_R).
\end{aligned}
\tag{4.65}
$$

The three first terms in (4.65) originate (in reverse order) from the trivial orbit (4.47), the rank one orbit (4.59), and the splitting degeneration contribution (4.60). By construction, the trivial orbit reproduces the two-loop contribution computed in (B.57). More remarkably, the rank one orbit matches the one-loop contribution (B.14), while the splitting degeneration contribution reproduces the tree-level $\nabla^2(\nabla\phi)^4$, obtained by dimensional reduction of the $\nabla^2 F^4$ coupling in 10 dimensions.[12]

The exponentially suppressed terms in the second line of (4.65) can be interpreted as instantons from Euclidean NS five-branes wrapped respectively on any possible $T^6$ inside $T^7$, KK (6,1)-branes wrapped with any $S^1$ Taub-NUT fiber in $T^7$, and H-monopoles wrapped on $T^7$. One has similarly for the other components (4.44)

$$
\begin{aligned}
g_3^6 G_{\alpha\beta,\gamma1}^{(2k,8),1,Q} &= \frac{3}{4i\sqrt{2Q_R^2}}e^{-\frac{2\pi}{g_3^2}\sqrt{2Q_R^2}}\bar{G}_{\langle\alpha\beta,}^{(2k-1,7)}(Q,\varphi)Q_{L\gamma\rangle}\,, \\
g_3^6 G_{\alpha\beta,11}^{(2k,8),1,Q} &= -\frac{1}{2\sqrt{2Q_R^2}}e^{-\frac{2\pi}{g_3^2}\sqrt{2Q_R^2}}\bar{G}_{\alpha\beta}^{(2k-1,7)}(Q,\varphi)\,,
\end{aligned}
\tag{4.66}
$$

where $\bar{G}_{\alpha\beta,-\frac{Q^2}{2}}^{(2k-1,7)} = \bar{G}_{F,\alpha\beta,-\frac{Q^2}{2}}^{(2k-1,7)} + \bar{G}_{P,\alpha\beta,-\frac{Q^2}{2}}^{(2k-1,7)}$ and takes the form

$$
\begin{aligned}
\bar{G}_{\alpha\beta,-\frac{Q^2}{2}}^{(2k-1,7)}(Q,\varphi) =\ & \sum_{\substack{d>0 \\ Q/d\in\Lambda_{2k-1,7}}} d^2 c_k\Big(-\frac{Q^2}{2d^2}\Big)G_{\alpha\beta,-\frac{Q^2}{2d^2}}^{(2k-1,7)}\Big(\frac{Q}{d}\Big) \\
& + \sum_{\substack{d>0 \\ Q/d\in N\Lambda_{2k-1,7}^*}} (Nd)^2 c_k\Big(-\frac{Q^2}{2Nd^2}\Big)^{\varsigma}G_{\alpha\beta,-\frac{Q^2}{2Nd^2}}^{(2k-1,7)}\Big(\frac{Q}{Nd}\Big).
\end{aligned}
\tag{4.67}
$$

---

[12]As already noted in [13], there also exists a tree-level single trace $\nabla^2 F^4$ interaction in ten dimensions, with coefficient proportional to $\zeta(3)$ [48], but the latter vanishes when all gauge bosons belong to an Abelian subalgebra and therefore does not contribute to the $\nabla^2(\nabla\phi)^4$ interaction in three dimensions. Note that the single trace interaction is not protected and receives corrections to all orders in heterotic perturbation theory [69].

For the null charges $Q^2 = 0$, we write instead the finite contribution as

$$\bar{G}_{F,\alpha\beta,0}^{(2k-1,7)}(Q,\varphi) = \frac{k}{N-1} \sum_{\substack{d>0 \\ Q/d \in \Lambda_{2k-1,7}}} d^2 \Big[ N\, {}^\varsigma G_{F,\alpha\beta,0}^{(2k-1,7)\perp}\big(\tfrac{Q}{d}\big) - G_{F,\alpha\beta,0}^{(2k-1,7)\perp}\big(\tfrac{Q}{d}\big) \Big]$$
$$+ \frac{k}{N-1} \sum_{\substack{d>0 \\ Q/d \in N\Lambda_{2k-1,7}^*}} (Nd)^2 \Big[ N G_{F,\alpha\beta,0}^{(2k-1,7)\perp}\big(\tfrac{Q}{Nd}\big) - {}^\varsigma G_{F,\alpha\beta,0}^{(2k-1,7)\perp}\big(\tfrac{Q}{Nd}\big) \Big]. \quad (4.68)$$

In the maximal rank case $N = 1$, upon setting ${}^\varsigma G_{ab}^{(p,q)} = G_{ab}^{(p,q)}$ and replacing $c_k(m) \to c(m)$, $k \to 12 = c(0)/2$, Eqs. (4.67) and (4.68) simplify to

$$\bar{G}_{\alpha\beta,-\frac{Q^2}{2}}^{(23,7)}(Q,\varphi) = \sum_{\substack{d>0 \\ Q/d \in \Lambda_{23,7}}} d^2\, c\big(-\tfrac{Q^2}{2d^2}\big) G_{\alpha\beta,-\frac{Q^2}{2d^2}}^{(23,7)}\big(\tfrac{Q}{d}\big). \quad (4.69)$$

It is important to note that the orbit method misses exponentially suppressed terms which do not depend on the axions $a$ in the last line of (4.65). The existence of these terms is clear from the differential constraint (3.20), since the $(\nabla\phi)^4$ coupling $F_{abcd}$ appearing on the right-hand side contains both instanton and anti-instanton contributions. Unfortunately, our current tools do not allow us to extract these contributions from the unfolding method at present. One could obtain them by solving the differential equation (E.51) for $Q = 0$.

Finally, it is worth stressing that while the perturbative contributions $G_{ab}^{(2k-1,7)}$ and $G_{ab,cd}^{(2k-1,7)}$ have singularities in codimension 7 inside $\mathcal{M}_3$ at points of enhanced gauge symmetry, the full instanton-corrected coupling (1.7) has only singularities in codimension 8. In Appendix B.3, we analyze the structure of the singularities for a general genus-two modular integral of the form (2.30) and find the expected one-loop and two-loop contributions with nearly massless gauge bosons running in the loops.

## 5  Large radius expansion of exact $\nabla^2(\nabla\phi)^4$ couplings

We now study the asymptotic expansion of the modular integral (1.7) in the limit where the radius $R$ of one circle in the internal space goes to infinity. We show that it reproduces the known $\nabla^2 F^4$ and $\mathcal{R}^2 F^2$ couplings in $D = 4$, along with an infinite series of $\mathcal{O}(e^{-R})$ corrections from 1/2-BPS and 1/4-BPS dyons whose wordline winds around the circle, up to an infinite series of $\mathcal{O}(e^{-R^2})$ corrections with non-zero NUT charge, corresponding to Taub-NUT instantons. We start by analyzing the expansion of genus-two modular integral (2.30) for arbitrary values of $(p,q)$, in the limit near the cusp where $O(p,q)$ is broken to $SL(2,\mathbb{R}) \times O(p-2,q-2)$, so that the moduli space decomposes into

$$G_{p,q} \to \mathbb{R}^+ \times \left[ \frac{SL(2,\mathbb{R})}{SO(2)} \times G_{p-2,q-2} \right] \ltimes \mathbb{R}^{2(p+q-4)} \times \mathbb{R}. \quad (5.1)$$

As in the previous section, we first discuss the maximal rank case $N = 1$, $p - q = 16$, where the integrand is invariant under the full modular group, before dealing with the case of $N$ prime. The reader uninterested by the details of the derivation may skip to §5.3, where we specialize to the values $(p,q) = (2k,8)$ relevant for the $\nabla^2(\nabla\phi)^4$ couplings in $D = 3$, and interpret the various contributions arising in the decompactification limit to $D = 4$. In §6 we generalize the results herein to degenerations of the form $O(p,q) \to SL(n) \times O(p-n,q-n)$, and apply these results to study weak coupling limits in type II and type I string vacua.

## 5.1 $O(p,q) \to O(p-2,q-2)$ for even self-dual lattices

In this subsection we assume that the lattice $\Lambda_{p,q}$ is even self-dual and factorizes in the limit (5.1) as

$$\Lambda_{p,q} \to \Lambda_{p-2,q-2} \oplus I\!I_{2,2} \,. \tag{5.2}$$

We denote by $R, t, a^{Ii}, \psi$ the coordinates for each factors in (5.1) (here $i = 1,2$ and $I = 3, \ldots, p+q-2$). The coordinate $R$ (not to be confused with the one used in §4) parametrizes a one-parameter subgroup $e^{RH_1}$ in $O(p,q)$, such that the action of the non-compact Cartan generator $H_1$ on the Lie algebra $\mathfrak{so}_{p,q}$ decomposes into

$$\mathfrak{so}_{p,q} \simeq \ldots \oplus (\mathfrak{gl}_1 \oplus \mathfrak{sl}_2 \oplus \mathfrak{so}_{p-2,q-2})^{(0)} \oplus (\mathbf{2} \otimes (\mathbf{p}+\mathbf{q}-\mathbf{4}))^{(1)} \oplus \mathbf{1}^{(2)}, \tag{5.3}$$

while $(a^{iI}, \psi)$ parametrize the unipotent subgroup obtained by exponentiating the grade 1 and 2 components in this decomposition. We parametrize the $SO(2)\backslash SL(2,\mathbb{R})$ coset representative $v_\mu{}^i$ and the symmetric $SL(2,\mathbb{R})$ element $M \equiv v^T v$ by the complex upper half-plane coordinate $S = S_1 + iS_2$, such that

$$v_\mu{}^i = \frac{1}{\sqrt{S_2}} \begin{pmatrix} 1 & S_1 \\ 0 & S_2 \end{pmatrix}, \quad M^{ij} = \delta^{\mu\nu} v_\mu{}^i v_\nu{}^j = \frac{1}{S_2} \begin{pmatrix} 1 & S_1 \\ S_1 & |S|^2 \end{pmatrix} \,. \tag{5.4}$$

The remaining coordinates in $G_{p-2,q-2}$ will be denoted by $\varphi$. As in the weak coupling expansion, lattice vector are labelled according to the choice of A-cycle on the genus-two Riemann surface. A generic charge vector $(Q^1_\mathcal{I}, Q^2_\mathcal{I}) \in \Lambda_{p,q} \oplus \Lambda_{p,q} \simeq (\mathbf{2} \otimes \mathbf{2})^{(-1)} \oplus (\mathbf{2} \otimes (\mathbf{p}+\mathbf{q}-\mathbf{4}))^{(0)} \oplus (\mathbf{2} \otimes \mathbf{2})^{(1)}$ decomposes into

$$(Q^1_\mathcal{I}, Q^2_\mathcal{I}) = (n^1_i, n^2_i, \widetilde{Q}^1_I, \widetilde{Q}^2_I, m^{1j}, m^{2j}), \tag{5.5}$$

where $(n^1_i, n^2_i, m^{1j}, m^{2j}) \in I\!I_{2,2} \oplus I\!I_{2,2}$ and $(\widetilde{Q}^1_I, \widetilde{Q}^2_I) \in \Lambda_{p-2,q-2} \oplus \Lambda_{p-2,q-2}$ such that $Q^r \cdot Q^s = -m^{ri} n^s_i - m^{si} n^r_i + \widetilde{Q}^r \cdot \widetilde{Q}^s$. The orthogonal projectors defined by $Q^r_L \equiv p^\mathcal{I}_L Q^r_\mathcal{I}$ and $Q^r_R \equiv p^\mathcal{I}_R Q^r_\mathcal{I}$ decompose according to

$$\begin{aligned}
p^\mathcal{I}_{L,\mu} Q^r_\mathcal{I} &= \frac{v^{-1}_{i\mu}}{R\sqrt{2}} \left( m^{ri} + a^i \cdot \widetilde{Q}^r + (\psi \varepsilon^{ij} + \tfrac{1}{2} a^i \cdot a^j) n^r_j \right) - \frac{R}{\sqrt{2}} v_\mu{}^i n^r_i, \\
p^\mathcal{I}_{L,\alpha} Q^r_\mathcal{I} &= \tilde{p}^I_{L,\alpha} (\widetilde{Q}^r_I + n^r_i a^i_I), \\
p^\mathcal{I}_{R,\mu} Q^r_\mathcal{I} &= \frac{v^{-1}_{i\mu}}{R\sqrt{2}} \left( m^{ri} + a^i \cdot \widetilde{Q}^r + (\psi \varepsilon^{ij} + \tfrac{1}{2} a^i \cdot a^j) n^r_j \right) + \frac{R}{\sqrt{2}} v_\mu{}^i n^r_i, \\
p^\mathcal{I}_{R,\hat{\alpha}} Q^r_\mathcal{I} &= \tilde{p}^I_{R,\hat{\alpha}} (\widetilde{Q}^r_I + n^r_i a^i_I),
\end{aligned} \tag{5.6}$$

where $\tilde{p}^I_{L,\alpha}, \tilde{p}^I_{R,\hat{\alpha}}$ ($\alpha = 3 \ldots p$, $\hat{\alpha} = 3 \ldots q$) are orthogonal projectors in $G_{p-2,q-2}$ satisfying $\widetilde{Q}^r \widetilde{Q}^s = \widetilde{Q}^r_L \cdot \widetilde{Q}^s_L - \widetilde{Q}^r_R \cdot \widetilde{Q}^s_R$.

In order to study the region $R \gg 1$ it is useful to perform a Poisson resummation on the momenta $m^{ri}$ along $I\!I_{2,2} \oplus I\!I_{2,2}$. Note that this analysis is in principle valid for a region containing $R > \sqrt{2}$. Insertion of momenta polynomials along the torus or the sublattice can be again obtained using an insertion of a auxiliary variables $(y_{r,\mu}, y_{r,\alpha})$

$$\begin{aligned}
&\Gamma^{(2)}_{\Lambda_{p,q}} \left[ e^{2\pi i y_a \cdot \widetilde{Q}^a + \frac{\pi}{2} y_a \cdot \Omega_2^{-1} \cdot y^a} \right] \\
&= R^4 \sum_{(\mathbf{m}_i, \mathbf{n}_j) \in \mathbb{Z}^8} e^{-\pi R^2 (\mathbf{n}_i \, \mathbf{m}_i) \binom{\Omega}{\mathbb{1}} \cdot \Omega_2^{-1} M^{ij} \cdot \left[ (\mathbf{n}_j \, \mathbf{m}_j) \binom{\bar{\Omega}}{\mathbb{1}} \right]^{\mathsf{T}}} e^{\frac{2\pi R}{i\sqrt{2}} y^\mu \cdot \Omega_2^{-1} \cdot \left[ (\mathbf{n}_i \, \mathbf{m}_i) \binom{\bar{\Omega}}{\mathbb{1}} \right]^{\mathsf{T}} v_\mu{}^i} \\
&\qquad \times \Gamma^{(2)}_{\Lambda_{p-2,q-2}} \left[ e^{2\pi i \mathbf{m}_i \cdot (a^i_I \widetilde{Q}^I + \frac{1}{2} a^i_I a^{Ij} \mathbf{n}_j)} e^{2\pi i y_{\alpha I} \cdot \widetilde{Q}^{\alpha I} + \frac{\pi}{2} y_{\alpha I} \cdot \Omega_2^{-1} \cdot y^{\alpha I}} \right], \tag{5.7}
\end{aligned}$$

where the sum over indices $r = 1, 2$ is implicit, we used Einstein summation convention for indices $r = 1, 2, \mu = 1, 2, i, j = 1, 2$ and $\alpha = 3, ..., p$, and where $M^{ij}$ is defined in (5.4). In this representation, modular invariance is manifest since a transformation $\Omega \mapsto (A\Omega + B)(C\Omega + D)^{-1}$ can be compensated by a linear transformation $\begin{pmatrix} \mathbf{n}_1 & \mathbf{m}_1 \\ \mathbf{n}_2 & \mathbf{m}_2 \end{pmatrix} \mapsto \begin{pmatrix} \mathbf{n}_1 & \mathbf{m}_1 \\ \mathbf{n}_2 & \mathbf{m}_2 \end{pmatrix} \begin{pmatrix} D^\mathsf{T} & -B^\mathsf{T} \\ -C^\mathsf{T} & A^\mathsf{T} \end{pmatrix}$, $y_\mu \mapsto y_\mu \cdot (C\Omega + D)$, under which the third line of (5.7) transforms as a weight $\frac{p-q}{2}$ modular form. We can therefore decompose charges $(\mathbf{n}_i, \mathbf{m}_j)$ into various orbits under $Sp(4, \mathbb{Z})$ and apply the unfolding trick to each orbit:

**The trivial orbit** $(\mathbf{n}_i, \mathbf{m}_j) = (0, 0)$ produces the integral (4.1) for the lattice $\Lambda_{p-2,q-2}^{\oplus 2} \equiv \Lambda_{p-2,q-2} \oplus \Lambda_{p-2,q-2}$, up to a factor $R^4$, and vanishes if one of the indices $ab, cd$ lies along $1, 2$

$$G_{\alpha\beta,\gamma\delta}^{(p,q),0} = R^4 \, G_{\alpha\beta,\gamma\delta}^{(p-2,q-2)} \,. \tag{5.8}$$

**Rank-one orbit** This orbit consists of matrices $(\mathbf{n}_i, \mathbf{m}_j) \neq (0, 0)$ where $(\mathbf{n}_1, \mathbf{m}_1)$ and $(\mathbf{n}_2, \mathbf{m}_2)$ are collinear and not simultaneously vanishing. Such matrices can be decomposed as $(\mathbf{n}_i, \mathbf{m}_j) = \binom{j}{p}(c_3, c_4, d_3, d_4)$, $(j, p) \neq (0, 0)$ and $\gcd(c_3, c_4, d_3, d_4) = 1$. Quadruplets $(c_3, c_4, d_3, d_4)$ with $\gcd(c_3, c_4, d_3, d_4) = 1$ can all be rotated to $(0, 0, 0, \pm 1)$ by a $Sp(4, \mathbb{Z})$ element, whose stabilizer is the central extension of the Jacobi group $\Gamma_1^J$ (4.10), and are in one-to-one correspondence with elements of $\Gamma_1^J \backslash Sp(4, \mathbb{Z})$. Thus for each doublet $(j, p) \neq (0, 0)$, one can unfold the integration domain $Sp(4, \mathbb{Z}) \backslash \mathcal{H}_2$ to $\Gamma_1^J \backslash \mathcal{H}_2 = \mathbb{R}_t^+ \times (SL(2, \mathbb{Z}) \backslash \mathcal{H}_1)_\rho \times (T^3 / \mathbb{Z}_2)_{u_1, u_2, \sigma_1}$ (for further details, see below (4.11)). We parametrize $\Gamma_1^J \backslash \mathcal{H}_2$ by $t = \frac{|\Omega_2|}{\rho_2}$, $\rho$ and $(u_1, u_2, \sigma_1) = (v_1 - u_2 \rho_1, v_2 / \rho_2, \sigma_1)$, and change the $y$ variables $(y_{1\mu}', y_{2\mu}', y_{1\alpha}', y_{2\alpha}') = (y_{1\mu}, y_{1\mu} u_1 - y_{2\mu}, y_{1\alpha}, y_{1\alpha} u_2 - y_{2\alpha})$ stabilizing $\mathcal{P}_{ab,cd}$

$$
\begin{aligned}
G_{ab,cd}^{(p,q),1} = R^4 \int_0^\infty \frac{dt}{t^3} \int_{\left[-\frac{1}{2}, \frac{1}{2}\right]^3} du_1 du_2 d\sigma_1 \int_{\mathcal{F}_1} \frac{d\rho_1 d\rho_2}{\rho_2^2} \frac{\mathcal{P}_{ab,cd}(\frac{\partial}{\partial y'})}{\Phi_{10}} {\sum_{(j,p) \in \mathbb{Z}^2}}' e^{-\frac{\pi R^2}{S_2 t}|j + pS|^2} \\
\times \Gamma_{\Lambda_{p-2,q-2}}^{(2)} \Big[ e^{2\pi i (j a_1^I + p a_2^I) \widetilde{Q}_{2I}} \exp 2\pi i \Big( \frac{R}{i\sqrt{2}} y_{r\mu}' (\Omega_2^{-1})^{r2} m_{2i} v^{i\mu} \\
+ y_{1\alpha}' (\widetilde{Q}_L^{1\alpha} + u_2 \widetilde{Q}_L^{2\alpha}) - y_{2\alpha}' \widetilde{Q}_L^{2\alpha} + \frac{1}{4i\rho_2} y_{1\alpha}' y_1'^\alpha + \frac{1}{4it} y_{2\alpha}' y_2'^\alpha \Big) \Big],
\end{aligned}
\tag{5.9}
$$

where $m_{2i} v^{i\mu} = \frac{1}{S_2} \begin{pmatrix} 1 & S_1 \\ 0 & S_2 \end{pmatrix} \binom{j}{p}$, and $\mathcal{P}_{ab,cd}(\frac{\partial}{\partial y})$ is derivative polynomial of order four defined in (4.13), and where the Fourier-Jacobi expansion of $1/\Phi_{10}$ is given eq.(4.14).

The integral over $\sigma_1$ picks up the Jacobi $\psi_m(\rho, v)$ of index $m = -\frac{1}{2} \widetilde{Q}_2^2$. Contributions from $\widetilde{Q}_2 = 0$ pick up the contribution $c(0) \widehat{E}_2 / (12\Delta)$ (4.15), and lead to power-like terms[13]

$$
\begin{aligned}
G_{\alpha\beta,\gamma\delta}^{(p,q),1,0} &= -R^{q-4} \frac{c(0)}{16\pi} \mathcal{E}^\star \big( \tfrac{8-q}{2}, S \big) \delta_{\langle \alpha\beta}, G_{\gamma\delta \rangle}^{(p-2,q-2)} \\
G_{\alpha\beta,\mu\nu}^{(p,q),1,0} &= -R^{q-4} \frac{c(0)}{48\pi} \Big[ \tfrac{8-q}{2} \delta_{\mu\nu} - 2 \mathcal{D}_{\mu\nu} \Big] \mathcal{E}^\star \big( \tfrac{8-q}{2}, S \big) G_{\alpha\beta}^{(p-1,q-1)},
\end{aligned}
\tag{5.10}
$$

where $\mathcal{E}^\star(s, S)$ is the completed weight 0 non-holomorphic Eisenstein series

$$\mathcal{E}^\star(s, S) = \frac{1}{2} \pi^{-s} \Gamma(s) {\sum_{(m,n) \in \mathbb{Z}^2}}' \frac{S_2^s}{|nS + m|^{2s}} \equiv \xi(2s) \mathcal{E}(s, S), \tag{5.11}$$

---

[13]Note that (5.10) has a pole at $q = 6$ and $q = 8$, of which the first is substracted by the regularization prescription discussed in §B.2.4, and the second cancels against the pole from the trivial orbit contribution (5.8).

with $\xi(2s)$ the reduced zeta function $\xi(2s) = \pi^{-s}\Gamma(s)\zeta(2s)$ and $\mathcal{D}_{\mu\nu}$ is the traceless differential operator on $\frac{SL(2,\mathbb{R})}{SO(2)}$ acting on $S$ and defined in terms of raising and lowering operators of weight $w$ as

$$\mathcal{D}_{\mu\nu} = -\frac{1}{2}\sigma^+_{\mu\nu}\mathcal{D}_w - \frac{1}{2}\sigma^-_{\mu\nu}\bar{\mathcal{D}}_w, \tag{5.12}$$

with $\sigma^\pm = \frac{1}{2}(\sigma_3 \pm i\sigma_1)$ and $\sigma_i$ the Pauli matrices.

Non-zero vectors $\widetilde{Q}_2$ lead to exponentially suppressed contributions, in a similar fashion as what described for the $O(p,q) \to O(p-1,q-1)$ limit, section 4.1. They depend on the axions through a phase factor $e^{2\pi i m_{2j}\widetilde{Q}_{2I}a^{Ij}}$. In order to evaluate them, we insert the Fourier-Jacobi expansion (A.54) and decompose each $\psi_m(\rho,v)$ into its finite and polar parts. In either case, the integral over $\sigma_1$ imposes $\widetilde{Q}_2^2 = -2m$. As in the previous section, we consider first the contributions of the finite part $\psi_m^F(\rho,v)$, for null and non-null vectors, and then the contributions of the polar part $\psi_m^P(\rho,v)$

1. In the case $\widetilde{Q}_2^2 = 0$, one can make the same decomposition as in section 4.1, using the constraint $\widetilde{Q}_1 \cdot \widetilde{Q}_2 = 0$ from $\widehat{\psi}_0^F(\rho)$. The integral then reads, for a given null vector $\widetilde{Q}_2$ and $m_{2j} = \binom{j}{p}$

$$
\begin{aligned}
\frac{R^4}{2} &\sideset{}{'}\sum_{(j,p)\in\mathbb{Z}^2} e^{2\pi i m_{2j}\widetilde{Q}_{2I}a^{Ij}} \gcd(\widetilde{Q}_2) \int_{\mathbb{R}^+} \frac{\mathrm{d}t}{t} t^{\frac{q-2}{2}} e^{-\frac{\pi R^2}{S_2 t}|j+pS|^2 - 2\pi t|\widetilde{Q}_{2R}|^2} \int_{\mathcal{F}_1} \frac{\mathrm{d}\rho_1\mathrm{d}\rho_2}{\rho_2^2}\frac{c(0)\widehat{E}_2}{12\Delta}\\
&\times \int_{\mathbb{R}} \mathrm{d}u_2\, \rho_2^{\frac{q-1}{2}} \sum_{\widetilde{Q}_1^\perp\in\Lambda_{p-3,q-3}} q^{\frac{1}{2}(\widetilde{Q}_1^\perp+u_2\widetilde{Q}_2)_L^2}\bar{q}^{\frac{1}{2}(\widetilde{Q}_1^\perp+u_2\widetilde{Q}_2)_R^2}\mathcal{P}_{ab,cd}\left(\tfrac{\partial}{\partial y'}\right)\\
&\times e^{2\pi i\left(\frac{R}{i\sqrt{2}}y'_{r\mu}(\Omega_2^{-1})^{r2}m_{2i}v^{i\mu} + y'_{1a}(\widetilde{Q}_1^\perp+u_2\widetilde{Q}_L^{1\alpha}) - y'_{2\alpha}\widetilde{Q}_L^{2\alpha} + \frac{1}{4i\rho_2}y'_{1a}y_1'^{\ \alpha} + \frac{1}{4it}y'_{2\alpha}y_2'^{\ \alpha}\right)}\bigg|_{y'=0},
\end{aligned}
\tag{5.13}
$$

where $\gcd(\widetilde{Q}_2)$ comes from unfolding the $u_2$-integral that uses the component of $\widetilde{Q}_1$ along $\widetilde{Q}_2$, and where $\widetilde{Q}_1^\perp \in \Lambda_{p-3,q-3}$ such that $\Lambda_{p-3,q-3} = \{\widetilde{Q}_1^\perp \in \Lambda_{p-2,q-2}, \widetilde{Q}_1^\perp \cdot \widetilde{Q}_2 = 0\}/$ $(\mathbb{Z}\frac{\widetilde{Q}_2}{\gcd\widetilde{Q}_2})$ (for further details, see (4.21)). We obtain the a one-loop integral on a sub-Grassmaniann $G_{p-2,q-2}$ parametrizing a space orthogonal to $\widetilde{Q}_2$, labelled $G_{F,\alpha\beta}^{(p-2,q-2)\perp}(\widetilde{Q}_2,\varphi)$, that we define as

$$
\begin{aligned}
G_{F,\alpha\beta,0}^{(p-2,q-2)\perp}(Q,\varphi) = \frac{\gcd(Q)}{12} \int_{\mathcal{F}_1} \frac{\mathrm{d}\rho_1\mathrm{d}\rho_2}{\rho_2^2}\frac{\widehat{E}_2}{\Delta_k(\rho)}\rho_2^{\frac{q-3}{2}} \sum_{\widetilde{Q}\in\Lambda_{p-3,q-3}} q^{\frac{1}{2}\widetilde{Q}_L^2}\bar{q}^{\frac{1}{2}\widetilde{Q}_R^2}e^{2\pi\rho_2\frac{(\widetilde{Q}_R\cdot Q_R)^2}{Q_R^2}}\\
\times\left[\left(\widetilde{Q}_{L\alpha} - \frac{\widetilde{Q}_R\cdot Q_R}{Q_R^2}Q_{L\alpha}\right)\left(\widetilde{Q}_{L\beta} - \frac{\widetilde{Q}_R\cdot Q_R}{Q_R^2}Q_{L\beta}\right) - \frac{1}{4\pi\rho_2}\left(\delta_{\alpha\beta} - \frac{Q_{L\alpha}Q_{L\beta}}{Q_R^2}\right)\right],
\end{aligned}
\tag{5.14}
$$

where $\Delta_k = \Delta$ in the case at hand. After defining $\Gamma_i = (Q,P) = m_{2i}\widetilde{Q}_2$, with support on 1/2-BPS states, and covariantizing the expression with the torus vielbein, we find that the Fourier coefficient with support $\Gamma_i \in \Lambda_{p-2,q-2}^{\oplus 2} \setminus \{0\}$, with $\varepsilon^{ij}\Gamma_i\Gamma_j = 0$, and mass $\mathcal{M}(\Gamma) = \sqrt{2M_{ij}\Gamma_R^i \cdot \Gamma_R^j}$, is given by, when $\widetilde{Q}_2^2 = \Gamma_i \cdot \Gamma_j = 0$

$$3R^{\frac{q+2}{2}}\bar{G}_{F,\langle\alpha\beta,0}^{(p-2,q-2)}(\Gamma,\varphi)\sum_{l=0}^1 \frac{\mathcal{P}_{\gamma\delta}^{(l)}(\Gamma,S)}{R^l}\frac{K_{\frac{q-6}{2}-l}(2\pi R\mathcal{M}(\Gamma))}{\mathcal{M}(\Gamma)^{\frac{q-4}{2}-l}}, \tag{5.15}$$

where the polynomial $\mathcal{P}^{(l)}$ in (4.23) is defined in appendix H.2, and

$$\bar{G}^{(p-2,q-2)}_{F,\alpha\beta,0}(\Gamma,\varphi) = c(0)\Big[\frac{1}{\sqrt{S_2}}|j'+p'S|\Big]^{q-8} \sum_{\substack{d\geq 1 \\ \hat{Q}/d\in\Lambda_{p-2,q-2}}} d^{q-8}\, G^{(p-2,q-2)\perp}_{F,\alpha\beta,0}(\tfrac{\hat{Q}}{d},\varphi)\,, \qquad (5.16)$$

and where we defined $\hat{Q}$ and the unique coprimes $(j',p')$ such that $\Gamma = (Q,P) = (j',p')\hat{Q}$. The full expression for all polarizations will be given together with the polar contributions in (5.22).

2. In the case $\widetilde{Q}_2^2 \neq 0$, we replace $\widehat{\psi}^F_m$ by its theta decomposition (4.25). The integral over $\sigma_1$ matches $\widetilde{Q}_2^2 = -2m$, while the integral over $u_1$ imposes the constraint $\widetilde{Q}_1\cdot\widetilde{Q}_2 = -\ell$. The variable $s\in\mathbb{Z}$ in (A.62) can be used to unfold the integral over $u_2\in[-\frac{1}{2},\frac{1}{2}]$ to $\mathbb{R}$, after shifting each term in the lattice sum as $\widetilde{Q}_1 \to \widetilde{Q}_1 + s\widetilde{Q}_2$, since $\widetilde{Q}_1,\widetilde{Q}_2 \in \Lambda_{p-2,q-2}$. One thus obtain a Fourier coefficient similar to previous case, using $\Gamma_i = m_{2i}\widetilde{Q}_2 = (Q,P)$,

$$3R^{\frac{q+2}{2}}\, \bar{G}^{(p-2,q-2)}_{F,\langle\alpha\beta,-\frac{\gcd(\Gamma_i\cdot\Gamma_j)}{2}}(\Gamma,\varphi) \sum_{l=0}^{1} \frac{\mathcal{P}^{(l)}_{\gamma\delta\rangle}(\Gamma,S)}{R^l} \frac{K_{\frac{q-6}{2}-l}(2\pi R\mathcal{M}(\Gamma))}{\mathcal{M}(\Gamma)^{\frac{q-4}{2}-l}}\,, \qquad (5.17)$$

where we denoted, by extension, the function

$$\bar{G}^{(p-2,q-2)}_{F,\alpha\beta,-\frac{\gcd(\Gamma_i\cdot\Gamma_j)}{2}}(\Gamma,\varphi) = \big(M^{ij}\Gamma_i\cdot\Gamma_j\big)^{\frac{q-8}{2}}$$
$$\times \sum_{\substack{d\geq 1 \\ \Gamma/d\in\Lambda^{\oplus 2}_{p-2,q-2}}} \Big(\frac{d^2}{\gcd(\Gamma_i\cdot\Gamma_j)}\Big)^{\frac{q-8}{2}} c\Big(-\frac{\gcd(\Gamma_i\cdot\Gamma_j)}{2d^2}\Big) G^{(p-2,q-2)\perp}_{F,\alpha\beta,-\frac{\gcd(\Gamma_i\cdot\Gamma_j)}{2d^2}}(\tfrac{\hat{Q}}{d},\varphi)\,, \qquad (5.18)$$

where we introduced the automorphic tensor $G^{(p-2,q-2)\perp}_{F,\alpha\beta,-\frac{\gcd(\Gamma_i\cdot\Gamma_j)}{2d^2}}(\tfrac{\hat{Q}}{d},\varphi)$ in (4.28) and the monomials $\mathcal{P}^{(l)}_{\gamma\delta}(\Gamma,S)$ in (H.2). Notice that the function $G^{(p-2,q-2)\perp}_{F,\alpha\beta,-\frac{\gcd(\Gamma_i\cdot\Gamma_j)}{2d^2}}(\tfrac{\hat{Q}}{d},\varphi)$ only depends on the direction of $\Gamma = (j',p')\hat{Q}$ in $\Lambda_{p-2,q-2}$, and on the norm $\gcd(\Gamma_i\cdot\Gamma_j)/d^2 = \hat{Q}^2/d^2$. The full expression for all polarizations will be given together with the polar contributions in (5.22).

3. For the polar contributions, we use the representation

$$\widehat{\psi}^P_m(\rho,\nu) = \frac{c(m)}{\Delta(\rho)} \sum_{s,\ell\in\mathbb{Z}} q^{ms^2+s\ell} y^{2ms+\ell}\, \hat{b}(s,\ell,m,\rho_2)\,. \qquad (5.19)$$

One can then shift the charges to $\widetilde{Q}_1 \to \widetilde{Q}_1 + s\widetilde{Q}_2$ since $\widetilde{Q}_1,\widetilde{Q}_2 \in \Lambda_{p-2,q-2}$, and then use the sum over $s$ to unfold the $u_2\in[-\frac{1}{2},\frac{1}{2}]$ to $\mathbb{R}$. Then, integrating over $u_1\in[-\frac{1}{2},\frac{1}{2}]$ imposes $\widetilde{Q}_1\cdot\widetilde{Q}_2 = -\ell$. One obtains the Fourier coefficients, using $\Gamma_i = m_{2i}\widetilde{Q}_2 = (Q,P)$,

$$3R^{\frac{q+2}{2}}\, \bar{G}^{(p-2,q-2)}_{P,\langle\alpha\beta,}(\Gamma,\varphi) \sum_{l=0}^{1} \frac{\mathcal{P}^{(l)}_{\gamma\delta\rangle}(\Gamma)}{R^l} \frac{K_{\frac{q-6}{2}-l}(2\pi R\mathcal{M}(\Gamma))}{\mathcal{M}(\Gamma)^{\frac{q-4}{2}-l}}\,, \qquad (5.20)$$

where

$$\bar{G}^{(p-2,q-2)}_{P,\alpha\beta}(\Gamma,\varphi) = \Big[\frac{1}{\sqrt{S_2}}|j'+p'S|\Big]^{q-8} \sum_{\substack{d\geq 1 \\ \hat{Q}/d\in\Lambda^{\oplus 2}_{p-2,q-2}}} c\Big(-\frac{\hat{Q}^2}{2d^2}\Big) d^{q-8} G^{(p-2,q-2)}_{P,\alpha\beta}(\tfrac{\hat{Q}}{d},\varphi)\,. \qquad (5.21)$$

Here $(j', p')$ are coprimes such that $\Gamma = (j', p')\hat{Q}$, and where we used the automorphic tensor $G^{(p,q)}_{P,ab}(\hat{Q}, \varphi)$ defined in (4.37). Note that the expression above is identical to (5.18), but expressed in a different manner to include the case where the norm of $\Gamma$ vanishes.

Combining all contributions, the sum of the finite and polar contributions to the rank one Fourier mode are given for all polarizations by

$$
\begin{aligned}
G^{(p,q),1,\Gamma}_{\alpha\beta,\gamma\delta} &= 3R^{\frac{q+2}{2}} \bar{G}^{(p-2,q-2)}_{\langle\alpha\beta,}(\Gamma, \varphi) \sum_{l=0}^{1} \frac{\mathcal{P}^{(l)}_{\gamma\delta\rangle}(\Gamma)}{R^l} \frac{K_{\frac{q-6}{2}-l}(2\pi R \mathcal{M}(\Gamma))}{\mathcal{M}(\Gamma)^{\frac{q-4}{2}-l}}, \\
G^{(p,q),1,\Gamma}_{\alpha\beta,\gamma\mu} &= \frac{3}{2} R^{\frac{q+2}{2}} \bar{G}^{(p-2,q-2)}_{\langle\alpha\beta,}(\Gamma, \varphi) \frac{\Gamma_{L\gamma\rangle\mu}}{i\sqrt{2}} \frac{K_{\frac{q-8}{2}}(2\pi R \mathcal{M}(\Gamma))}{\mathcal{M}(\Gamma)^{\frac{q-6}{2}}}, \\
G^{(p,q),1,\Gamma}_{\alpha\beta,\mu\nu} &= -R^{\frac{q+2}{2}} \bar{G}^{(p-2,q-2)}_{\alpha\beta}(\Gamma, \varphi) \Gamma_{R\hat{\alpha}\mu} \Gamma_R{}^{\hat{\alpha}}{}_\nu \frac{K_{\frac{q-10}{2}}(2\pi R \mathcal{M}(\Gamma))}{\mathcal{M}(\Gamma)^{\frac{q-4}{2}}},
\end{aligned}
\tag{5.22}
$$

where $\bar{G}^{(p-2,q-2)}_{\alpha\beta}(\Gamma, \varphi) = \bar{G}^{(p-2,q-2)}_{F,\alpha\beta,-\frac{\gcd(\Gamma_i \cdot \Gamma_j)}{2}}(\Gamma, \varphi) + \bar{G}^{(p-2,q-2)}_{P,\alpha\beta}(\Gamma, \varphi)$, $\Gamma_{L\gamma\mu} = v_\mu{}^i \Gamma_{L\gamma i}$, $\Gamma_{R\hat{\alpha}\mu} = v_\mu{}^i \Gamma_{R\hat{\alpha}i}$, and we recall $\Gamma_i = (Q, P)$.

**Rank two Abelian orbits**  These orbits consist of matrices $\begin{pmatrix} \mathbf{n}_1 & \mathbf{m}_1 \\ \mathbf{n}_2 & \mathbf{m}_2 \end{pmatrix}$ where $(\mathbf{n}_1, \mathbf{m}_1)$ and $(\mathbf{n}_2, \mathbf{m}_2)$ are not collinear (in particular, non-zero) but have vanishing symplectic product $\mathbf{n}_1 \cdot \mathbf{m}_2 - \mathbf{m}_1 \cdot \mathbf{n}_2 = 0$. Such matrices can be decomposed as $\begin{pmatrix} \mathbf{n}_1 & \mathbf{m}_1 \\ \mathbf{n}_2 & \mathbf{m}_2 \end{pmatrix} = \begin{pmatrix} 0 & \mathbf{j} \\ 0 & \mathbf{p} \end{pmatrix} \begin{pmatrix} A & B \\ C & D \end{pmatrix}$, where $(\mathbf{j}, \mathbf{p}) \in M_2(\mathbb{Z}) \backslash \{0\}$, and $\begin{pmatrix} A & B \\ C & D \end{pmatrix} \in \Gamma_{2,\infty} \backslash Sp(4, \mathbb{Z})$, with $\Gamma_{2,\infty} = GL(2, \mathbb{Z}) \ltimes \mathbb{Z}^3$ the residual symmetry at the cusp $\Omega_2 \to \infty$, embedded in $Sp(4, \mathbb{Z})$ as

$$
\Gamma_{2,\infty} = \{ \begin{pmatrix} \gamma & 0 \\ 0 & \gamma^{-\intercal} \end{pmatrix}, \gamma \in GL(2, \mathbb{Z}) \} \ltimes \{ \begin{pmatrix} \mathbb{1} & M \\ 0 & \mathbb{1} \end{pmatrix}, M \in M_2(\mathbb{Z}), M = M^\intercal \}.
\tag{5.23}
$$

Doublets $(C, D)$ can be rotated to $(0, \mathbb{1})$ by an element of $Sp(4, \mathbb{Z})$, and are in one-to-one correspondence with elements $\Gamma_{2,\infty} \backslash Sp(4, \mathbb{Z})$. The fundamental domain can thus be unfolded from $Sp(4, \mathbb{Z}) \backslash \mathcal{H}_2$ to $\Gamma_{2,\infty} \backslash \mathcal{H}_2 = (GL(2, \mathbb{Z}) \backslash \mathcal{P}_2)_{\Omega_2} \ltimes (\mathbb{R}/\mathbb{Z})^3_{\Omega_1}$, where $\mathcal{P}_2$ is the set of positive-definite matrices. Finally, one can restrict the matrices $A = (\mathbf{j}, \mathbf{p}) \in M_2(\mathbb{Z})$ to $A \in M_2(\mathbb{Z})/GL(2, \mathbb{Z})$, in order to unfold $GL(2, \mathbb{Z}) \backslash \mathcal{P}_2$ to $\mathcal{P}_2$.

The resulting contribution can be expressed in terms of the auxiliary variables $(y_{r,\mu}, y_{r,\alpha})$ (5.7), and we obtain

$$
\begin{aligned}
G^{(p,q),2\text{Ab}}_{ab,cd} = &\ 2R^4 \int_{\mathcal{P}_2} \frac{d^3\Omega_2}{|\Omega_2|^3} \int_{[-\frac{1}{2},\frac{1}{2}]^3} d^3\Omega_1 \frac{|\Omega_2|^{\frac{q-2}{2}}}{\Phi_{10}} \\
&\ \times \sum_{\tilde{Q} \in \Lambda^{\oplus 2}_{p-2,q-2}} e^{\pi i \text{Tr}[\Omega \tilde{Q} \cdot \tilde{Q}^\intercal]} \sum_{A \in M_2(\mathbb{Z})/GL(2,\mathbb{Z})} e^{2\pi i a^{iI} A_{ij} Q_I^j - \pi \text{Tr}[\frac{R^2}{S_2} \Omega_2^{-1} A^\intercal \begin{pmatrix} 1 & S_1 \\ S_1 & |S|^2 \end{pmatrix} A + 2\Omega_2 \tilde{Q}_R \cdot \tilde{Q}_R^\intercal]} \\
&\ \times \mathcal{P}_{ab,cd}\left(\frac{\partial}{\partial y}\right) e^{2\pi i \left(\frac{R}{i\sqrt{2}} y_{r\mu}(\Omega_2^{-1})^{rs} A_{si}^\intercal v^{\intercal i\mu} + y_{r\alpha} \tilde{Q}_L{}^{r\alpha} + \frac{1}{4i} y_{r\alpha}(\Omega_2^{-1})^{rs} y_s{}^\alpha\right)},
\end{aligned}
\tag{5.24}
$$

where the factor two comes from the non-trivial center of order 2 of $GL(2, \mathbb{Z})$ acting on $\mathcal{H}_2$. For sufficiently large $|\Omega_2|$, the integral over $\Omega_1 \in [0, 1]^3$ selects the Fourier coefficient $C(m, n, L; \Omega_2)$ of $1/\Phi_{10}$, with $\tilde{Q}_1^2 = -2m$, $\tilde{Q}_2^2 = -2n$, $\tilde{Q}_1 \cdot \tilde{Q}_2 = -L$. As discussed in §A.6, the Fourier coefficient can be decomposed into a finite contribution $C^F(n, m, L)$, independent of

$\Omega_2$, and an infinite series of terms associated to the polar part,

$$
\begin{aligned}
C(m,n,L;\Omega_2) &\equiv \int_{\mathcal{C}} d^3\Omega_1 \frac{e^{i\pi(Q_1,Q_2)\begin{pmatrix} \rho & \nu \\ \nu & \sigma \end{pmatrix}\begin{pmatrix} Q_1 \\ Q_2 \end{pmatrix}}}{\Phi_{10}} \\
&= C^F(Q_1^2, Q_2^2, Q_1 \cdot Q_2) \\
&+ \sum_{\gamma \in GL(2,\mathbb{Z})/\mathrm{Dih}_4} c\left(-\tfrac{(sQ_1 - qQ_2)^2}{2}\right) c\left(-\tfrac{(pQ_2 - rQ_1)^2}{2}\right)\Big[ -\frac{\delta\big(\mathrm{tr}\big(\big(\begin{smallmatrix} 0 & 1/2 \\ 1/2 & 0 \end{smallmatrix}\big)\gamma^{\intercal}\Omega_2\gamma\big)\big)}{4\pi} \\
&+ \tfrac{(sQ_1 - qQ_2)\cdot(pQ_2 - rQ_1)}{2}\big(\mathrm{sign}\big((sQ_1 - qQ_2)\cdot(pQ_2 - rQ_1)\big) - \mathrm{sign}\big(\mathrm{tr}\big(\big(\begin{smallmatrix} 0 & 1/2 \\ 1/2 & 0 \end{smallmatrix}\big)\gamma^{\intercal}\Omega_2\gamma\big)\big)\big)\Big],
\end{aligned}
$$
(5.25)

where $\gamma = \begin{pmatrix} p & q \\ r & s \end{pmatrix}$ and $\mathrm{Dih}_4 \equiv \big\langle \big(\begin{smallmatrix} 1 & 0 \\ 0 & -1 \end{smallmatrix}\big), \big(\begin{smallmatrix} 0 & 1 \\ 1 & 0 \end{smallmatrix}\big)\big\rangle$ is the dihedral group of order 8, which stabilizes (up to sign) the matrix $\big(\begin{smallmatrix} 0 & 1/2 \\ 1/2 & 0 \end{smallmatrix}\big)$, or equivalently the locus $\nu_2 = 0$. As explained in Appendix A.6, this formula holds only when $|\Omega_2| > 1/4$, such that the contour $\mathcal{C} = [0,1]^3 + i\Omega_2$ avoids the poles of $1/\Phi_{10}$ for generic values of $\Omega_2$. Inserting (5.25) in (5.24), we find the following contributions,

1. The contributions from $(\widetilde{Q}_1, \widetilde{Q}_2) = (0,0)$ produces power-like terms in $R^2$, from the delta function contribution in (5.25), even though $C^F(0,0,0) = 0$,

$$
\begin{aligned}
G^{(p,q),\,2\mathrm{Ab},0}_{\alpha\beta,\gamma\delta} &= -R^{2q-12}\frac{3c(0)^2}{64\pi^3}\mathcal{E}^{\star}(\tfrac{8-q}{2},S)^2 \delta_{\langle\alpha\beta}\delta_{\gamma\delta\rangle}\,, \\
G^{(p,q),\,2\mathrm{Ab},0}_{\alpha\beta,\rho\sigma} &= -R^{2q-12}\frac{c(0)^2}{32\pi^3}\mathcal{E}^{\star}(\tfrac{8-q}{2},S)\big[\tfrac{8-q}{2}\delta_{\rho\sigma} - 2\mathcal{D}_{\rho\sigma}\big]\mathcal{E}^{\star}(\tfrac{8-q}{2},S)\,\delta_{\alpha\beta}\,, \\
G^{(p,q),\,2\mathrm{Ab},0}_{\mu\nu,\rho\sigma} &= -R^{2q-12}\frac{3c(0)^2}{64\pi^3}\big[\tfrac{8-q}{2}\delta_{\langle\mu\nu,} - 2\mathcal{D}_{\langle\mu\nu,}\big]\mathcal{E}^{\star}(\tfrac{8-q}{2},S)\big[\tfrac{8-q}{2}\delta_{\rho\sigma\rangle} - 2\mathcal{D}_{\rho\sigma\rangle}\big]\mathcal{E}^{\star}(\tfrac{8-q}{2},S)\,.
\end{aligned}
$$
(5.26)

   Here, the non-holomorphic Eisenstein series $\mathcal{E}^{\star}(s,S)$ and traceless differential operator $\mathcal{D}_{\mu\nu}$ are defined in (5.11) and (5.12). It is worth noting that in the limit $S_2 \to \infty$, the constant term proportional to $\xi(q-6)S_2^{\frac{q-6}{2}}$ in the Eisenstein series $\mathcal{E}^{\star}(\tfrac{8-q}{2},S)$ reproduces the missing constant term in (4.20). Thus, while this term is missed by the unfolding procedure in the degeneration $(p,q) \to (p-1,q-1)$, it is correctly captured by the unfolding procedure in the degeneration $(p,q) \to (p-2,q-2)$.

2. Contributions of non-zero vectors $(\widetilde{Q}_1, \widetilde{Q}_2) \in \Lambda^{\oplus 2}_{p-2,q-2}$ lead to exponentially suppressed contributions. For the finite term $C^F(Q_1^2, Q_2^2, Q_1 \cdot Q_2)$ in (5.25), and for the simplest tensorial representation, the unfolded integral leads to

$$
6R^8 \delta_{\langle\mu\nu,}\delta_{\rho\sigma\rangle} \sum_{\substack{(\widetilde{Q}_1,\widetilde{Q}_2)\in\Lambda^{\oplus 2}_{p-2,q-2} \\ A\in M_2(\mathbb{Z})/GL(2,\mathbb{Z})}} |A|^2\, e^{2\pi i a^{iI}A_{ij}Q_I^j}\, C^F(Q_1^2, Q_2^2, Q_1\cdot Q_2)\Big(\tfrac{R^2|A|}{2|\widetilde{Q}_{1R}\wedge\widetilde{Q}_{2R}|}\Big)^{\frac{q-9}{2}}\widetilde{B}_{\frac{q-9}{2}}(Z)\,,
$$
(5.27)

   where

$$
Z = \frac{2R^2}{S_2}\begin{pmatrix} 1 & S_1 \\ S_1 & |S|^2 \end{pmatrix}A\begin{pmatrix} \widetilde{Q}_{1R}^2 & \widetilde{Q}_{1R}\cdot\widetilde{Q}_{2R} \\ \widetilde{Q}_{1R}\cdot\widetilde{Q}_{2R} & \widetilde{Q}_{2R}^2 \end{pmatrix}A^{\intercal}\,,
$$
(5.28)

   $|Q \wedge P|^2 = \det\big(\begin{smallmatrix} Q^2 & Q\cdot P \\ Q\cdot P & P^2 \end{smallmatrix}\big)$ and $\widetilde{B}_{\delta}(Z)$ is the matrix-variate Bessel function [70], defined by

$$
\widetilde{B}_s(UV) = \tfrac{1}{2}\Big(\frac{|U|}{|V|}\Big)^{-s/2}\int_{\mathcal{P}_2}\frac{d^3\Omega_2}{|\Omega_2|^{\frac{3}{2}-s}}e^{-\pi\,\mathrm{tr}(\Omega_2^{-1}U + \Omega_2 V)}\,.
$$
(5.29)

Note that $\widetilde{B}_\delta(Z)$ depends on $Z$ only through its trace and determinant. In the limit $R \to \infty$, or large $|Z| = |UV|$, the integral over $\Omega_2$ is dominated by a saddle point where $\Omega_2^\star V \Omega_2^\star = U$; using the identity $\mathrm{Tr}(UV)U - UVU = |U||V|V^{-1}$ valid for $2 \times 2$ matrices, this is given by

$$\Omega_2^\star = \frac{U + \sqrt{|UV|}\,V^{-1}}{\sqrt{\mathrm{tr}(UV) + 2\sqrt{|UV|}}}\,. \tag{5.30}$$

For the matrices $U = \frac{R^2}{S_2}A^\intercal \left(\begin{smallmatrix} 1 & S_1 \\ S_1 & |S|^2 \end{smallmatrix}\right)A$, $V = 2\left(\begin{smallmatrix} \widetilde{Q}_{1R}^2 & \widetilde{Q}_{1R}\cdot\widetilde{Q}_{2R} \\ \widetilde{Q}_{1R}\cdot\widetilde{Q}_{2R} & \widetilde{Q}_{2R}^2 \end{smallmatrix}\right)$, given by (5.24), we obtain

$$\Omega_2^\star = \frac{R}{\mathcal{M}(\Gamma)}A^\intercal\left(\frac{1}{\sqrt{S_2}}\left(\begin{smallmatrix} 1 & S_1 \\ S_1 & |S|^2 \end{smallmatrix}\right) + \frac{1}{|Q_R \wedge P_R|}\left(\begin{smallmatrix} P_R^2 & -Q_R \cdot P_R \\ -Q_R \cdot P_R & Q_R^2 \end{smallmatrix}\right)\right)A, \tag{5.31}$$

where $\mathcal{M}(\Gamma)$ is the mass (2.5) of a 1/4-BPS state with charge $\Gamma = (Q, P) = (\widetilde{Q}_1, \widetilde{Q}_2)A^\intercal$, and $|Q_R \wedge P_R| = \sqrt{Q_R^2 P_R^2 - (Q_R \cdot P_R)^2}$.

For the contributions on the last line of (5.25), the integral over $\Omega_2$ no longer evaluates to a matrix-variate Bessel integral, since these contributions depend on $\Omega_2$, being discontinuous across the walls where $\mathrm{tr}\left(\left(\begin{smallmatrix} 0 & 1/2 \\ 1/2 & 0 \end{smallmatrix}\right)\gamma^\intercal \Omega_2 \gamma\right)$ changes sign. However, as long as (5.31) does not sit on the walls, the integral over $\Omega_2$ is still dominated by the same saddle point, with a prefactor obtained by replacing $C^F(Q_1^2, Q_2^2, Q_1 \cdot Q_2)$ by $C(Q_1^2, Q_2^2, Q_1 \cdot Q_2; \Omega_2^*)$. In appendix F, we estimate the error made by neglecting the variation of $C(Q_1^2, Q_2^2, Q_1 \cdot Q_2; \Omega_2)$ at finite distance away from the saddle point, and find that they are of the order expected for multi-instanton corrections. For the remainder of this section, we ignore these corrections, and perform the above replacement in (5.27).

In order to write the result for more general polarizations, it will be useful to introduce

$$\begin{aligned}
\widetilde{B}_{s,\mu\nu}^{(0)}(Z) &= \frac{\delta_{\mu\nu}}{4|Z|^{s/2}}\int \frac{\mathrm{d}^3\Omega_2}{|\Omega_2|^{\frac{3}{2}-s}}e^{-\pi\,\mathrm{tr}(\Omega_2^{-1}Z+\Omega_2)}, \\
\widetilde{B}_{s,\mu\nu}^{(1)}(Z) &= \frac{1}{2|Z|^{s/2}}\int \frac{\mathrm{d}^3\Omega_2}{|\Omega_2|^{1-s}}(\Omega_2^{-1})_{\mu\nu}\,e^{-\pi\,\mathrm{tr}(\Omega_2^{-1}Z+\Omega_2)},
\end{aligned} \tag{5.32}$$

such that $\delta^{\mu\nu}\widetilde{B}_{s\,\mu\nu}^{(0)}(Z) = \widetilde{B}_s(Z)$ and $|Z|^{\frac{s}{2}}\widetilde{B}_{s\,\mu\nu}^{(1)}(Z) = \frac{1}{-\pi}\frac{\partial}{\partial Z^{\mu\nu}}\left[\sqrt{|Z|}^{s+\frac{1}{2}}\widetilde{B}_{s+\frac{1}{2}}(Z)\right]$.

Changing variable $\left(\begin{smallmatrix} Q \\ P \end{smallmatrix}\right) = A\left(\begin{smallmatrix} \widetilde{Q}_1 \\ \widetilde{Q}_2 \end{smallmatrix}\right)$, we therefore obtain the Fourier expansion with respect to $(a_1, a_2)$, with support on $\Gamma = (Q, P) \in \Lambda_{p-2,q-2}^{\oplus 2}$,

$$\begin{aligned}
G_{\alpha\beta,\gamma\delta}^{(p,q),\,2\mathrm{Ab},\Gamma} &\sim 2R^{q-1}\bar{C}(Q,P;\Omega_2^\star)\sum_{l=0}^{2}\frac{P_{\alpha\beta,\gamma\delta}^{(l)\mu\nu}(\Gamma)}{R^l}\frac{\widetilde{B}_{\frac{q-5-l}{2},\mu\nu}^{(l\bmod 2)}\left[\frac{2R^2}{S_2}\left(\begin{smallmatrix} 1 & S_1 \\ S_1 & |S|^2 \end{smallmatrix}\right)\left(\begin{smallmatrix} Q_R^2 & Q_R \cdot P_R \\ Q_R \cdot P_R & P_R^2 \end{smallmatrix}\right)\right]}{|2Q_R \wedge P_R|^{\frac{q-5-l}{2}}}, \\
G_{\rho\beta,\gamma\delta}^{(p,q),\,2\mathrm{Ab},\Gamma} &\sim R^{q-1}\bar{C}(Q,P;\Omega_2^\star)\sum_{l=0}^{1}\frac{P_{\rho\beta,\gamma\delta}^{(l)\mu\nu}(\Gamma)}{R^l}\frac{\widetilde{B}_{\frac{q-6-l}{2},\mu\nu}^{(l+1\bmod 2)}\left[\frac{2R^2}{S_2}\left(\begin{smallmatrix} 1 & S_1 \\ S_1 & |S|^2 \end{smallmatrix}\right)\left(\begin{smallmatrix} Q_R^2 & Q_R \cdot P_R \\ Q_R \cdot P_R & P_R^2 \end{smallmatrix}\right)\right]}{|2Q_R \wedge P_R|^{\frac{q-6-l}{2}}}, \\
&\ \vdots \\
G_{\mu\nu,\rho\sigma}^{(p,q),\,2\mathrm{Ab},\Gamma} &\sim 2R^{q-1}\bar{C}(Q,P;\Omega_2^\star)\frac{\delta_{\langle\mu\nu,}\delta_{\sigma\tau\rangle}}{4}\frac{\widetilde{B}_{\frac{q-9}{2}}\left[\frac{2R^2}{S_2}\left(\begin{smallmatrix} 1 & S_1 \\ S_1 & |S|^2 \end{smallmatrix}\right)\left(\begin{smallmatrix} Q_R^2 & Q_R \cdot P_R \\ Q_R \cdot P_R & P_R^2 \end{smallmatrix}\right)\right]}{|2Q_R \wedge P_R|^{\frac{q-9}{2}}},
\end{aligned} \tag{5.33}$$

where the measure factor is given by, for $\Gamma = (Q, P)$

$$\bar{C}(Q,P;\Omega_2^\star) = \sum_{\substack{A \in M_2(\mathbb{Z})/GL(2,\mathbb{Z}) \\ A^{-1}\Gamma \in \Lambda_{p-2,q-2}^{\oplus 2}}} |A|^{q-7}C\left[A^{-1}\left(\begin{smallmatrix} Q^2 & Q \cdot P \\ Q \cdot P & P^2 \end{smallmatrix}\right)A^{-\intercal}; A^\intercal \Omega_2^\star A\right]. \tag{5.34}$$

3. Contributions from the Dirac delta function and sign function in the first line of (5.25) also produce exponentially suppressed contributions to the same Fourier coefficient. These contributions are localized on the walls $\mathrm{tr}\left(\begin{pmatrix} 0 & 1/2 \\ 1/2 & 0 \end{pmatrix}\gamma^{\mathsf{T}}\Omega_2\gamma\right)$ associated to the splittings $(Q,P) = (Q_1,P_1)+(Q_2,P_2)$. For the Dirac delta function terms the integral separates into the product of two Bessel functions, with arguments given by the masses $\mathcal{M}(Q_1,P_1)$ and $\mathcal{M}(Q_2,P_2)$ of the 1/2-BPS components, as shown in Appendix D. In Appendix C, we show that he summation measure for these contributions also factorizes into the two respective measures for 1/2-BPS instantons appearing in the genus-one integral (1.4), (1.6). The contributions from the sign functions are estimated in Appendix F.

**Rank two non-abelian orbits**   These orbits consist of matrices $\begin{pmatrix} \mathbf{n}_1 & \mathbf{m}_1 \\ \mathbf{n}_2 & \mathbf{m}_2 \end{pmatrix}$ where $(\mathbf{n}_1,\mathbf{m}_1)$ and $(\mathbf{n}_2,\mathbf{m}_2)$ have non vanishing symplectic product $M_1 \equiv \mathbf{n}_1 \cdot \mathbf{m}_2 - \mathbf{m}_1 \cdot \mathbf{n}_2 \neq 0$ (in particular, they are non collinear). Unlike all other orbits considered previously, the contribution of such matrices depend on the scalar $\psi$ corresponding to the top grade component in the decomposition (5.3) via a factor $e^{2i\pi M_1\psi}$, and therefore contribute to the non-Abelian Fourier coefficient. While the classification of the orbits of such matrices under $Sp(4,\mathbb{Z})$ is rather complicated, we show in Appendix G that these contributions can be deduced by a simple change of variables from the already known Fourier coefficients in the degeneration $(p,q) \to (p-1,q-1)$.

## 5.2   Extension to $\mathbb{Z}_N$ CHL orbifolds

The degeneration limit (5.1) of the modular integral (2.30) for $\mathbb{Z}_N$ CHL models with $N = 2,3,5,7$ can be treated similarly by adapting the orbit method to the case where the integrand is invariant under the congruence subgroup $\Gamma_{2,0}(N) = \{\begin{pmatrix} A & B \\ C & D \end{pmatrix} \in Sp(4,\mathbb{Z}), C = 0 \bmod N\}$. In (1.7), $\Phi_{k-2}$ is the cusp form of $\Gamma_{2,0}(N)$ of weight $k = \frac{24}{N+1}$ defined in (A.33), and $\Gamma^{(2)}_{\Lambda_{p,q}}[P_{ab,cd}]$ is the genus-two partition function with insertion of $P_{ab,cd}$ for a lattice

$$\Lambda_{p,q} = \Lambda_{p-2,q-2} \oplus I\!I_{1,1} \oplus I\!I_{1,1}[N]\,, \tag{5.35}$$

where $\Lambda_{p-2,q-2}$ is a lattice of level $N$ with signature $(p-2,q-2)$. The lattice $I\!I_{1,1} \oplus I\!I_{1,1}[N]$ is obtained from the usual unimodular lattice $I\!I_{2,2}$ by restricting the windings and momenta to $\begin{pmatrix} \mathbf{n}_1 & \mathbf{m}_1 \\ \mathbf{n}_2 & \mathbf{m}_2 \end{pmatrix} = \begin{pmatrix} n_{11} & n_{12} & m_{11} & m_{12} \\ n_{21} & n_{22} & m_{21} & m_{22} \end{pmatrix} \in \begin{pmatrix} \mathbb{Z}^2 & \mathbb{Z}^2 \\ (N\mathbb{Z})^2 & \mathbb{Z}^2 \end{pmatrix}$, hence breaking the automorphism group $O(2,2,\mathbb{Z})$ to $\sigma_{S\leftrightarrow T} \ltimes [\Gamma_0(N) \times \Gamma_0(N)]$, exactly as in [22]. After Poisson resummation on $\mathbf{m}_1,\mathbf{m}_2$, Eq. (4.7) continues to hold, except for the fact that $\mathbf{n}_2$ are restricted to run over $(N\mathbb{Z})^2$. The sum over $A = \begin{pmatrix} \mathbf{n}_1 & \mathbf{m}_1 \\ \mathbf{n}_2 & \mathbf{m}_2 \end{pmatrix}$ can then be decomposed into orbits of $\Gamma_{2,0}(N)$:

**Trivial orbit**   The term $\begin{pmatrix} \mathbf{n}_1 & \mathbf{m}_1 \\ \mathbf{n}_2 & \mathbf{m}_2 \end{pmatrix} = \begin{pmatrix} 0 & 0 \\ 0 & 0 \end{pmatrix}$ produces the same modular integral, up to a factor of $R^4$,

$$G^{(p,q),0}_{\alpha\beta,\gamma\delta} = R^4\, G^{(p-2,q-2)}_{\alpha\beta,\gamma\delta}\,, \tag{5.36}$$

where $G^{(p-1,q-1)}_{\alpha\beta,\gamma\delta}$ is the integral (4.1) for the lattice $\Lambda_{p-2,q-2}$ defined by (5.35).

**Rank-one orbits**   Matrices $A$ of rank one fall into two different classes of orbits under $\Gamma_{2,0}(N)$. Let us first consider the case where $(\mathbf{n}_2,\mathbf{m}_2) \neq (0,0)$ and denote $(\mathbf{n}_2,\mathbf{m}_2) = p(\mathbf{n}'_2,\mathbf{m}'_2)$ with $p = \gcd(\mathbf{n}_2,\mathbf{m}_2)$:

1. Matrices with $\mathbf{n}'_2 = 0 \bmod N$, as they are required to be rank one, can be decomposed as

$$\begin{pmatrix} \mathbf{n}_1 & \mathbf{m}_1 \\ \mathbf{n}_2 & \mathbf{m}_2 \end{pmatrix} = \begin{pmatrix} 0 & 0 & 0 & j \\ 0 & 0 & 0 & p \end{pmatrix}\begin{pmatrix} A & B \\ C & D \end{pmatrix}, \tag{5.37}$$

with $(j,p) \in \mathbb{Z}^2 \smallsetminus \{(0,0)\}$, $p \neq 0$, and $\begin{pmatrix} A & B \\ C & D \end{pmatrix} \in (\Gamma_0(N) \ltimes H_{2,1}(\mathbb{Z}))\backslash\Gamma_{2,0}(N)$, with $\mathbb{Z}_2 \times \Gamma_0(N) \ltimes H_{2,1}(N) \subset \Gamma_1^J$. For this class of orbits, one can thus unfold directly the domain $\Gamma_{2,0}(N)\backslash\mathcal{H}_2$ into $(\Gamma_0(N) \ltimes H_{2,1}(\mathbb{Z}))\backslash\mathcal{H}_2 = \mathbb{R}_t^+ \times (\Gamma_0(N)\backslash\mathcal{H}_1)_\rho \times ((\mathbb{R}/\mathbb{Z})^3/\mathbb{Z}_2)_{u_1,u_2,\sigma_1}$ (for further details, see (4.48));

2. Matrices with $\mathbf{n}_2' \neq 0 \bmod N$ can be decomposed as

$$\begin{pmatrix} \mathbf{n}_1 & \mathbf{m}_1 \\ \mathbf{n}_2 & \mathbf{m}_2 \end{pmatrix} = \begin{pmatrix} 0 & j & 0 & 0 \\ 0 & p & 0 & 0 \end{pmatrix} \begin{pmatrix} A & B \\ C & D \end{pmatrix}, \tag{5.38}$$

with $(j,p) \in \mathbb{Z} \oplus N\mathbb{Z} \smallsetminus \{(0,0)\}$, $p \neq 0$, since $\mathbf{n}_2 = 0 \bmod N$, and where $\begin{pmatrix} A & B \\ C & D \end{pmatrix} \in S_\rho S_\sigma(\Gamma^0(N) \ltimes H_{2,1,N}^{(2)}(\mathbb{Z}))(S_\rho S_\sigma)^{-1}\backslash\Gamma_{2,0}(N)$, recalling the definition

$$H_{2,1,N}^{(2)}(\mathbb{Z}) = \{(\kappa,\lambda,\mu) \in H_{2,1}(\mathbb{Z}), \kappa = \mu = 0 \bmod N\}, \tag{5.39}$$

and where $S_\sigma$ denotes the inversion over $\sigma$. One can then unfold the fundamental domain $\Gamma_{2,0}(N)\backslash\mathcal{H}_2$ into $S_\rho S_\sigma (\Gamma^0(N) \ltimes H_{2,1,N}^{(2)}(\mathbb{Z}))(S_\rho S_\sigma)^{-1}\backslash\mathcal{H}_2$, and change variable $\Omega \to (S_\rho S_\sigma) \cdot \Omega = -\Omega^{-1}$ as in the weak coupling case (4.53) to recover the integration domain $(\Gamma^0(N) \ltimes H_{2,1,N}^{(2)}(\mathbb{Z}))\backslash\mathcal{H}_2 = \mathbb{R}_t^+ \times (\Gamma^0(N)\backslash\mathcal{H}_1)_\rho \times (\mathbb{R}/\mathbb{Z})_{u_2} \times (\mathbb{R}/N\mathbb{Z})^2_{u_1,\sigma_1}$. Under this change of variable, the level-$N$ weight-$(k-2)$ cusp form transforms as in (4.51), while the partition function for the sublattice $\Lambda_{p-2,q-2}$ transforms as

$$\Gamma^{(2)}_{\tilde{\Lambda}_{p-2,q-2}}[P_{\alpha\beta,\gamma\delta}](-\Omega^{-1}) = \upsilon^2 N^{-k-2}(-\mathrm{i})^{2k}|\Omega|^{k-2}\Gamma^{(2)}_{\tilde{\Lambda}^*_{p-2,q-2}}[P_{\alpha\beta,\gamma\delta}](\Omega), \tag{5.40}$$

where we denoted $\upsilon^2 N^{-k-2} = \left|\Lambda^*_{p-2,q-2}/\Lambda_{p-2,q-2}\right|^{-1}$ (Note that $\upsilon^2 = N^{2-2\delta_{q,8}}$ for $q \leq 8$ in the cases of interest).

The remaining contributions $A$ with $(\mathbf{n}_2, \mathbf{m}_2) = (0,0)$ can be split in the two classes of orbits above. Given $(\mathbf{n}_1, \mathbf{m}_1) = j(\mathbf{n}_1', \mathbf{m}_1')$, where $j = \gcd(\mathbf{n}_1, \mathbf{m}_1)$ and $j \in \mathbb{Z}$, terms with $\mathbf{n}_1' = 0 \bmod N$ correspond to cases $(j,p) = (j,0)$ in the first class above, while terms with $\mathbf{n}_1' \neq 0 \bmod N$ correspond to $(j,p) = (j,0)$ in the second class above.

For the function $G_{ab,cd}^{(p,q),1}$, changing the $y$ variables as before $(y'_{1\mu}, y'_{2\mu}, y'_{1\alpha}, y'_{2\alpha}) = (y_{1\mu}, y_{1\mu}u_1 - y_{2\mu}, y_{1\alpha}, y_{1\alpha}u_2 - y_{2\alpha})$, the sum of the two classes of orbits then reads (similarly to (4.53))

$$\begin{aligned}
G_{ab,cd}^{(p,q),1} = R^4 &\int_{\mathbb{R}^+} \frac{\mathrm{d}t}{t^3} \int_{(\mathbb{R}/\mathbb{Z})^3} \mathrm{d}u_1 \mathrm{d}u_2 \mathrm{d}\sigma_1 \int_{\Gamma_0(N)\backslash\mathcal{H}_1} \frac{\mathrm{d}\rho_1 \mathrm{d}\rho_2}{\rho_2^2} \frac{\mathcal{P}_{ab,cd}(\frac{\partial}{\partial y'})}{\Phi_{k-2}(\Omega)} \\
&\times \sum_{(j,p)\in\mathbb{Z}^2}{}' e^{-\frac{\pi R^2}{S_2 t}|j+pS|^2} \Gamma^{(2)}_{\Lambda_{p-2,q-2}}\left[e^{2\pi \mathrm{i}\tilde{Q}_{2I}(ja_1^I + pa_2^I)}\mathcal{Y}(y')\right] \\
+ R^4 &\int_{\mathbb{R}^+} \frac{\mathrm{d}t}{t^3} \int_{(\mathbb{R}/N\mathbb{Z})^2} \mathrm{d}u_1 \mathrm{d}\sigma_1 \int_{\mathbb{R}/\mathbb{Z}} \mathrm{d}u_2 \int_{\Gamma^0(N)\backslash\mathcal{H}_1} \frac{\mathrm{d}\rho_1 \mathrm{d}\rho_2}{\rho_2^2} \frac{\mathcal{P}_{ab,cd}(\frac{\partial}{\partial y'})}{\Phi_{k-2}(\Omega/N)} \\
&\times \frac{\upsilon^2}{N^4} \sum_{\substack{(j,p)\in\mathbb{Z}^2 \\ p=0\bmod N}}{}' e^{-\frac{\pi R^2}{S_2 t}|j+pS|^2} \Gamma^{(2)}_{\Lambda^*_{p-2,q-2}}\left[e^{2\pi \mathrm{i}\tilde{Q}_{2I}(ja_1^I + pa_2^I)}\mathcal{Y}(y')\right],
\end{aligned} \tag{5.41}$$

where

$$\mathcal{Y}(y') = e^{2\pi\mathrm{i}\left(\frac{R}{\mathrm{i}\sqrt{2}}\frac{m_i v^i_\mu y'^\mu_2}{t} + y'_{1\alpha}(Q_{1L}{}^\alpha + u_2 Q_{2L}{}^\alpha) - y'_{2\alpha}Q_{2L}{}^\alpha + \frac{1}{4\mathrm{i}\rho_2}y'_{1\alpha}y'^\alpha_1 + \frac{1}{4\mathrm{i}t}y'_{2\alpha}y'^\alpha_2\right)}, \tag{5.42}$$

with $m_i v^i_\mu = \frac{1}{\sqrt{S_2}}(j + p S_1, p S_2)$. The contributions with $\widetilde{Q}^2_2 = 0$, after integration over $u_1, u_2$ (4.55), can be brought back to regular integral over $\Gamma_0(N)\backslash\mathcal{H}_1$ by changing variable $\rho \to -1/\rho$. Similarly to (4.56), the transformation property of the genus-one partition function and the level-$N$ cusp form allows to obtain[14]

$$
G^{(p,q),1,Q^2=0}_{ab,cd} = R^4 \int_0^\infty \frac{\mathrm{d}t}{t} t^{\frac{q-6}{2}} \sum_{\substack{\widetilde{Q}_2 \in \tilde{\Lambda}_{p-2,q-2} \\ \widetilde{Q}^2_2 = 0}} e^{-2\pi t \widetilde{Q}^2_{2R}} \frac{c_k(0)}{12(N-1)}
$$

$$
\times \Bigg\{ {\sum_{(j,p)\in\mathbb{Z}^2}}' e^{-\frac{\pi R^2}{S_2 t}|j+pS|^2} \int_{\Gamma_0(N)\backslash\mathcal{H}_1} \frac{\mathrm{d}\rho_1 \mathrm{d}\rho_2}{\rho_2^2} \frac{N^2 \widehat{E}_2(N\rho) - \widehat{E}_2(\rho)}{\Delta_k}
$$

$$
+ \upsilon N {\sum_{\substack{(j,p)\in\mathbb{Z}^2 \\ p = 0 \bmod N}}}' e^{-\frac{\pi R^2}{S_2 t}|j+pS|^2} \int_{\Gamma_0(N)\backslash\mathcal{H}_1} \frac{\mathrm{d}\rho_1 \mathrm{d}\rho_2}{\rho_2^2} \frac{\widehat{E}_2(\rho) - \widehat{E}_2(N\rho)}{\Delta_k(\rho)} \Bigg\}
$$

$$
\times e^{2\pi i \widetilde{Q}_{2l}(ja_1^l + pa_2^l)} \Gamma_{\Lambda_{p-1,q-1}} \Big[ \mathcal{P}_{ab,cd}(\tfrac{\partial}{\partial y'}) \mathcal{Y}(y') \Big]. \tag{5.43}
$$

The zero mode contribution, $\widetilde{Q}_2 = 0$, lead to power-like terms

$$
G^{(p,q),1,0}_{\alpha\beta,\gamma\delta} = R^{q-4} \frac{c_k(0)}{16\pi(N-1)} \Big[ \delta_{\langle\alpha\beta,} G^{(p-2,q-2)}_{\gamma\delta\rangle}(\mathcal{E}^\star(\tfrac{8-q}{2},S) - \upsilon N^{\frac{q-6}{2}}\mathcal{E}^\star(\tfrac{8-q}{2},NS))
$$

$$
- \delta_{\langle\alpha\beta,}\, {}^\varsigma G^{(p-2,q-2)}_{\gamma\delta\rangle}(N\mathcal{E}^\star(\tfrac{8-q}{2},S) - \upsilon N^{\frac{q-8}{2}}\mathcal{E}^\star(\tfrac{8-q}{2},NS)) \Big],
$$

$$
G^{(p,q),1,0}_{\alpha\beta,\mu\nu} = R^{q-4} \frac{c_k(0)}{48\pi(N-1)} \Big[ \tfrac{8-q}{2}\delta_{\mu\nu} - 2\mathcal{D}_{\mu\nu} \Big]
$$

$$
\times \Big[ G^{(p-2,q-2)}_{\gamma\delta}(\mathcal{E}^\star(\tfrac{8-q}{2},S) - \upsilon N^{\frac{q-6}{2}}\mathcal{E}^\star(\tfrac{8-q}{2},NS))
$$

$$
- {}^\varsigma G^{(p-2,q-2)}_{\gamma\delta}(N\mathcal{E}^\star(\tfrac{8-q}{2},S) - \upsilon N^{\frac{q-8}{2}}\mathcal{E}^\star(\tfrac{8-q}{2},NS)) \Big], \tag{5.44}
$$

where we use the genus-one modular integral $G^{(p,q)}_{ab}(\varphi)$ (B.11), with integrand invariant under the Hecke congruence subgroup $\Gamma_0(N)$, as well as ${}^\varsigma G^{(p,q)}_{ab}$ (4.57) (Note that the cases of interest satisfy ${}^\varsigma G^{(2k-2,6)}_{ab}(\varphi) = G^{(2k-2,6)}_{ab}(\varphi)$, ${}^\varsigma G^{(2k-4,4)}_{ab}(\varphi) = G^{(2k-4,4)}_{ab}(\varphi)$).

The terms with non-zero vectors $Q$ lead to exponentially suppressed contributions of the same form as the Fourier modes of null vectors (5.15), non-null vectors (5.17), and the polar contribution (5.20) respectively, with the following changes:

1. In the case of the finite part of $1/\Phi_k(\Omega)$, for null Fourier vectors $Q^2 = 0$, the CHL equivalent of $G^{(p-2,q-2)}_{F,\alpha\beta,0}$ is

$$
G^{(p-2,q-2)}_{F\,\alpha\beta,0}(\Gamma_i,S) = \frac{c_k(0)}{12(N-1)} \Big[ \frac{|j'+p'S|}{\sqrt{S_2}} \Big]^{q-8}
$$

$$
\times \Bigg[ \Bigg( \sum_{\substack{d\geq 1 \\ \Gamma/d \in \Lambda^{\oplus 2}_{p-2,q-2}}} d^{q-8} - \sum_{\substack{d\geq 1 \\ \Gamma/d \in \Lambda^*_{p-2,q-2} \oplus N\Lambda^*_{p-2,q-2}}} \upsilon N d^{q-8} \Bigg) G^{(p-2,q-2)\perp}_{F,\alpha\beta,0}(\tfrac{\hat{Q}}{d},\varphi)
$$

$$
+ \Bigg( \sum_{\substack{d\geq 1 \\ \Gamma/d \in \Lambda^*_{p-2,q-2} \oplus N\Lambda^*_{p-2,q-2}}} \upsilon d^{q-8} - \sum_{\substack{d\geq 1 \\ \Gamma/d \in \Lambda^{\oplus 2}_{p-2,q-2}}} N d^{q-8} \Bigg) {}^\varsigma G^{(p-2,q-2)\perp}_{F,\alpha\beta,0}(\tfrac{\hat{Q}}{d},\varphi) \Bigg], \tag{5.45}
$$

where we defined the coprimes $(j', p')$ such that $\Gamma = (j', p')\hat{\Gamma}$.

---

[14]Recall that $\Delta_k(-1/N\rho) = N^{\frac{k}{2}}(-i\rho)^k \Delta_k(\rho)$, $\Gamma_{\Lambda^*_{p-2,q-2}}[P_{ab}](-1/\rho) = v^{-1} N^{\frac{k}{2}+1}(-i)^k \rho^{k-2} \Gamma_{\Lambda_{p-2,q-2}}[P_{ab}](\rho)$

2. For non-null Fourier vectors, $Q^2 \neq 0$, the finite part of $1/\Phi_{k-2}(\Omega)$ contains two terms

$$
\begin{aligned}
G^{(p-2,q-2)}_{\alpha\beta, -\frac{\gcd(\Gamma_i \cdot \Gamma_j)}{2d^2}}(\Gamma, \varphi) = \left(M^{ij}\Gamma_i \cdot \Gamma_j\right)^{\frac{q-8}{2}} \\
\Bigg( \sum_{\substack{d \geq 1 \\ \Gamma/d \in \Lambda^{\oplus 2}_{p-2,q-2}}} c_k\left(-\frac{\gcd(\Gamma_i \cdot \Gamma_j)}{2d^2}\right)\left(\frac{d^2}{\gcd(\Gamma_i \cdot \Gamma_j)}\right)^{\frac{q-8}{2}} G^{(p-2,q-2)\perp}_{\alpha\beta, -\frac{\gcd(\Gamma_i \cdot \Gamma_j)}{2d^2}}\left(\frac{\hat{Q}}{d}, \varphi\right) \\
+ \sum_{\substack{d \geq 1 \\ \Gamma/d \in \Lambda^*_{p-2,q-2} \oplus N\Lambda^*_{p-2,q-2}}} \upsilon\, c_k\left(-\frac{\gcd(\Gamma_i \cdot \Gamma_j)}{2Nd^2}\right)\left(\frac{Nd^2}{\gcd(\Gamma_i \cdot \Gamma_j)}\right)^{\frac{q-8}{2}} \varsigma G^{(p-2,q-2)\perp}_{\alpha\beta, -\frac{\gcd(\Gamma_i \cdot \Gamma_j)}{2Nd^2}}\left(\frac{\hat{Q}}{Nd}, \varphi\right) \Bigg),
\end{aligned}
\tag{5.46}
$$

with

$$
\varsigma G^{(p-2,q-2)\perp}_{\alpha\beta, m}\left(\frac{\Gamma}{Nd}, \varphi\right) = \int_{\Gamma_0(N)\backslash\mathcal{H}_1} \frac{d^2\rho}{\rho_2^2} \frac{N\widehat{h}_{m,l}(N\rho)}{\Delta_k(\rho)} \Gamma^{m,l}_{\alpha\beta}\left(\frac{\Gamma}{Nd}\right),
\tag{5.47}
$$

and $\Gamma^{m,l}_{ab}(Q)$ the vector-valued partition function defined in (4.29).

**Rank two abelian orbits**  Matrices $\begin{pmatrix} \mathbf{n}_1 & \mathbf{m}_1 \\ \mathbf{n}_2 & \mathbf{m}_2 \end{pmatrix} = \begin{pmatrix} n_{11} & n_{12} & m_{11} & m_{12} \\ n_{21} & n_{22} & m_{21} & m_{22} \end{pmatrix}$ with vanishing symplectic product $\mathbf{n}_1 \cdot \mathbf{m}_2 - \mathbf{m}_1 \cdot \mathbf{n}_2 = 0$ but $(\mathbf{n}_i, \mathbf{m}_i) \neq (0,0)$ and $(\mathbf{n}_1, \mathbf{m}_1)$ and $(\mathbf{n}_2, \mathbf{m}_2)$ not aligned, fall into four different classes of orbits. Consider $k_1 = \gcd(\mathbf{n}_1, \mathbf{m}_1)$ and $k_2 = \gcd(\mathbf{n}_2, \mathbf{m}_2)$, the four classes depend on whether $\mathbf{n}_1/k_1$ and $\mathbf{n}_2/k_2$ are congruent to $0 \bmod N$ or not.

1. When $\mathbf{n}_1/k_1$ and $\mathbf{n}_2/k_2 = 0 \bmod N$, one can rotate the element as $\begin{pmatrix} \mathbf{n}_1 & \mathbf{m}_1 \\ \mathbf{n}_2 & \mathbf{m}_2 \end{pmatrix} = \begin{pmatrix} 0 & \mathbf{p}_1 \\ 0 & \mathbf{p}_2 \end{pmatrix}\begin{pmatrix} A & B \\ C & D \end{pmatrix}$, with $(\mathbf{p}_1, \mathbf{p}_2) \in M_2(\mathbb{Z}) \smallsetminus \{0\}$ and $\begin{pmatrix} A & B \\ C & D \end{pmatrix} \in \Gamma_{2,\infty} \backslash \Gamma_{2,0}(N)$ ($A$, $C$ are not independent and the fourth winding entry, say $n_{22}$, vanishes because of the symplectic contraint). The representative is stabilized by $\Gamma_{2,\infty} = GL(2,\mathbb{Z}) \times T^3$, and one can restrict the sum over matrices $A = (\mathbf{j}, \mathbf{p}) \in M_2(\mathbb{Z})$ to $A \in M_2(\mathbb{Z})/GL(2,\mathbb{Z})$ and unfold the fundamental domain from $\Gamma_{2,0}(N)\backslash\mathcal{H}_2$ to $\Gamma_{2,\infty}\backslash\mathcal{H}_2 = (\mathcal{P}_2)_{\Omega_2} \times (\mathbb{R}\backslash\mathbb{Z})^3_{\Omega_1}$, with $(Q, P) \in \Lambda_m \oplus \Lambda_m$.

2. The two cases $\mathbf{n}_1/k_1 \neq 0 \bmod N$ but $\mathbf{n}_2/k_2 = 0 \bmod N$, and $\mathbf{n}_1/k_1 = 0 \bmod N$ but $\mathbf{n}_2/k_2 \neq 0 \bmod N$, should be considered together. Respectively, the charges can be rotated as $\begin{pmatrix} \mathbf{n}_1 & \mathbf{m}_1 \\ \mathbf{n}_2 & \mathbf{m}_2 \end{pmatrix} = \begin{pmatrix} k & 0 & 0 & j \\ 0 & 0 & 0 & p \end{pmatrix}\begin{pmatrix} A & B \\ C & D \end{pmatrix}, 0 \leq j < k, p \in \mathbb{Z}\smallsetminus\{0\}$, and $\begin{pmatrix} \mathbf{n}_1 & \mathbf{m}_1 \\ \mathbf{n}_2 & \mathbf{m}_2 \end{pmatrix} = \begin{pmatrix} 0 & j & k & 0 \\ 0 & p & 0 & 0 \end{pmatrix}\begin{pmatrix} A & B \\ C & D \end{pmatrix}$, $0 \leq j < Nk, p \in N\mathbb{Z}\smallsetminus\{0\}$, by construction of the lattice (5.35). $\begin{pmatrix} A & B \\ C & D \end{pmatrix} \in S_\rho \Gamma^{(1)}_{2,\infty,N} S_\rho^{-1}\backslash\Gamma_{2,0}(N)$ and $\begin{pmatrix} A & B \\ C & D \end{pmatrix} \in S_\sigma \Gamma^{(2)}_{2,\infty,N} S_\sigma^{-1}\backslash\Gamma_{2,0}(N)$ respectively, with

$$
\begin{aligned}
\Gamma^{(1)}_{2,\infty,N} &= \{\begin{pmatrix} \mathbb{1} & M \\ 0 & \mathbb{1} \end{pmatrix}, M = \begin{pmatrix} Nq & r \\ r & s \end{pmatrix}, (q,r,s) \in \mathbb{Z}^3\}, \\
\Gamma^{(2)}_{2,\infty,N} &= \{\begin{pmatrix} \mathbb{1} & M \\ 0 & \mathbb{1} \end{pmatrix}, M = \begin{pmatrix} q & r \\ r & Ns \end{pmatrix}, (q,r,s) \in \mathbb{Z}^3\},
\end{aligned}
\tag{5.48}
$$

and one can then unfold $\Gamma_{2,0}(N)\backslash\mathcal{H}_2$ to $S_\rho \Gamma^{(1)}_{2,\infty,N} S_\rho^{-1}\backslash\mathcal{H}_2$, $S_\sigma \Gamma^{(2)}_{2,\infty,N} S_\sigma^{-1}\backslash\mathcal{H}_2$, and change variable $\rho \to -1/\rho$, $\sigma \to -1/\sigma$, respectively. After exchanging $\rho$ and $\sigma$ in the second case[15], the two cases can be assembled together to form the two orbits of the decomposition of

$$
M_{2,0}(N) = \{\begin{pmatrix} p & q \\ r & s \end{pmatrix} \in M_2(\mathbb{Z}), r = 0 \bmod N\},
\tag{5.49}
$$

over

$$
(\mathbb{Z}_2 \ltimes \Gamma_0(N)) = \{\begin{pmatrix} p & q \\ r & s \end{pmatrix} \in GL(2,\mathbb{Z}), r = 0 \bmod N\}.
\tag{5.50}
$$

---

[15]This transformation belongs to $\Gamma_{2,0}(N)$

Explicitly,

$$M_{2,0}(N)/(\mathbb{Z}_2 \ltimes \Gamma_0(N)) = \{\begin{pmatrix} k & j \\ 0 & p \end{pmatrix}, 0 \le j < k, p \in \mathbb{Z} \smallsetminus \{0\}\}$$
$$\cup \{\begin{pmatrix} j & k \\ p & 0 \end{pmatrix}, 0 \le j < Nk, p \in N\mathbb{Z} \smallsetminus \{0\}\}.$$
(5.51)

One thus obtains a single sum over matrices $A \in M_{2,0}(N)/(\mathbb{Z}_2 \ltimes \Gamma_0(N))$, with a fundamental domain unfolded to $\Gamma_{2,\infty,N}^{(1)} \backslash \mathcal{H}_2 = (\mathcal{P}_2)_{\Omega_2} \times (\mathbb{R}\backslash\mathbb{Z})_{\sigma_1,v_1}^2 \times (\mathbb{R}\backslash N\mathbb{Z})_{\rho_1}$, with $(Q, P) \in \Lambda_m^* \oplus \Lambda_m$. Under this change of variable, the level-$N$ weight-$(k-2)$ cusp form transforms as

$$\Phi_{k-2}(S_\rho \circ \Omega) = (i\sqrt{N})^{-k} \rho^{k-2} \tilde{\Phi}_{k-2}(\Omega),$$
(5.52)

such that it satifies the splitting degeneration limit (A.44), while the genus-two partition function for the sublattice transforms as

$$\Gamma_{\Lambda_{p-2,q-2}}^{(2)}[P_{ab,cd}](S_\rho \circ \Omega) = v(i\sqrt{N})^{-k-2} \rho^{k-2} \Gamma_{\Lambda_{p-2,q-2}^* \oplus \Lambda_{p-2,q-2}}^{(2)}[P_{ab,cd}](\Omega),$$
(5.53)

where $v = N^{k/2+1}|\Lambda_{p-2,q-2}^*/\Lambda_{p-2,q-2}|^{-1/2}$ (reducing to $v = N^{1-\delta_{q,8}}$ for $q \le 8$ in the cases of interest).

3. When $\mathbf{n}_1/k_1, \mathbf{n}_2/k_2 \ne 0 \mod N$, one can rotate the element as $\begin{pmatrix} \mathbf{n}_1 & \mathbf{m}_1 \\ \mathbf{n}_2 & \mathbf{m}_2 \end{pmatrix} = \begin{pmatrix} j_1 & j_2 & 0 & 0 \\ p_1 & p_2 & 0 & 0 \end{pmatrix}\begin{pmatrix} A & B \\ C & D \end{pmatrix}$, with $\begin{pmatrix} j_1 & j_2 \\ p_1 & p_2 \end{pmatrix} \in M_{2,00}(N) \smallsetminus \{0\}$,

$$M_{2,00}(N) = \{\begin{pmatrix} p & q \\ r & s \end{pmatrix} \in M_2(\mathbb{Z}), r = s = 0 \mod N\}$$
(5.54)

by construction of the lattice (5.35), and $\begin{pmatrix} A & B \\ C & D \end{pmatrix} \in S_\rho S_\sigma \Gamma_{2,\infty,N}^{(3)}(S_\sigma S_\rho)^{-1} \backslash \Gamma_{2,0}(N)$, with

$$\Gamma_{2,\infty,N}^{(3)} = \{\begin{pmatrix} \gamma & 0 \\ 0 & \gamma^{-\intercal} \end{pmatrix}, \gamma \in GL(2, \mathbb{Z})\} \times \{\begin{pmatrix} \mathbb{1} & M \\ 0 & \mathbb{1} \end{pmatrix}, M = \begin{pmatrix} q & r \\ r & s \end{pmatrix}, (q, r, s) \in (N\mathbb{Z})^3\}.$$
(5.55)

One can then unfold $\Gamma_{2,0}(N)\backslash\mathcal{H}_2$ to $S_\rho S_\sigma \Gamma_{2,\infty,N}^{(3)}(S_\sigma S_\rho)^{-1}\backslash\mathcal{H}_2$, and change variable $\Omega_2 \to -\Omega_2^{-1}$ to recover $\Gamma_{2,\infty,N}^{(3)}\backslash\mathcal{H}_2 = (GL(2, \mathbb{Z})\backslash\mathcal{P}_2)_{\Omega_2} \times (\mathbb{R}\backslash N\mathbb{Z})_{\Omega_1}^3$. Finally, one can restrict the sum over matrices $A \in M_{2,00}(N)$, $\mathbf{p} = 0 \mod N$ to $A \in M_{2,00}(N)/GL(2, \mathbb{Z})$, in order to unfold $GL(2, \mathbb{Z})\backslash\mathcal{P}_2$ to $\mathcal{P}_2$, with $(Q, P) \in \Lambda_m^* \oplus N\Lambda_m^*$.

After unfolding and changing variables, the result for the simplest component $G_{\alpha\beta,\gamma\delta}^{(p,q),2\mathrm{Ab}}$ reads

$$G_{\alpha\beta,\gamma\delta}^{(p,q),2\mathrm{Ab}} = 2R^4 \int_{\mathcal{P}_2} \frac{\mathrm{d}^3\Omega_2}{|\Omega_2|^3} \int_{(\mathbb{R}/\mathbb{Z})^3} \frac{\mathrm{d}^3\Omega_1}{\Phi_{k-2}(\Omega)} \sum_{\substack{A \in \\ M_2(\mathbb{Z})/GL(2,\mathbb{Z})}}' e^{-\pi\mathrm{Tr}\left[\frac{R^2}{S_2}\Omega_2^{-1}A^\intercal\begin{pmatrix} 1 & S_1 \\ S_1 & |S|^2 \end{pmatrix}A\right]}$$
$$\times \Gamma_{\Lambda_{p-2,q-2}}^{(2)}[e^{2\pi\mathrm{i}a^{iI}A_{ij}\tilde{Q}^{jI}}P_{\alpha\beta,\gamma\delta}]$$

$$+2R^4 \int_{\mathcal{P}_2} \frac{\mathrm{d}^3\Omega_2}{|\Omega_2|^3} \int_{(\mathbb{R}/\mathbb{Z})^2\times(\mathbb{R}/N\mathbb{Z})} \frac{\mathrm{d}^3\Omega_1}{\tilde{\Phi}_{k-2}(\Omega)} \frac{v}{N} \sum_{\substack{A \in \\ M_{2,0}(N)/(\mathbb{Z}_2\times\Gamma_0(N))}}' e^{-\pi\mathrm{Tr}\left[\frac{R^2}{S_2}\Omega_2^{-1}A^\intercal\begin{pmatrix} 1 & S_1 \\ S_1 & |S|^2 \end{pmatrix}A\right]}$$
$$\times \Gamma_{\Lambda_{p-2,q-2}^* \oplus \Lambda_{p-2,q-2}}^{(2)}[e^{2\pi\mathrm{i}a^{iI}A_{ij}Q^{jI}}P_{\alpha\beta,\gamma\delta}]$$

$$+2R^4 \int_{\mathcal{P}_2} \frac{\mathrm{d}^3\Omega_2}{|\Omega_2|^3} \int_{(\mathbb{R}/N\mathbb{Z})^3} \frac{\mathrm{d}^3\Omega_1}{\Phi_{k-2}(\Omega/N)} \frac{v^2}{N^4} \sum_{\substack{A \in \\ M_{2,00}(N)/GL(2,\mathbb{Z})}}' e^{-\pi\mathrm{Tr}\left[\frac{R^2}{S_2}\Omega_2^{-1}A^\intercal\begin{pmatrix} 1 & S_1 \\ S_1 & |S|^2 \end{pmatrix}A\right]}$$
$$\times \Gamma_{\Lambda_{p-2,q-2}^*}^{(2)}[e^{2\pi\mathrm{i}a^{iI}A_{ij}Q^{jI}}P_{\alpha\beta,\gamma\delta}],$$
(5.56)

where $v^2 = N^{k+2}|\Lambda^*_{p-2,q-2}\backslash\Lambda_{p-2,q-2}|^{-1}$ (which reduces to $v^2 = N^{2-2\delta_{q,8}}$ for $q \leq 8$ in the cases of interest).

Integrating over $\Omega_1$ selects the Fourier coefficient $C_{k-2}(m,n,l;\Omega_2)$ of $1/\Phi_{k-2}$, and the Fourier coefficient $\widetilde{C}_{k-2}(m,n,l;\Omega_2)$ of $1/\widetilde{\Phi}_{k-2}$, with $\widetilde{Q}_1^2 = -2m$, $\widetilde{Q}_2 = -2n$, $\widetilde{Q}_1 \cdot \widetilde{Q}_2 = -l$. The first one is invariant under $GL(2,\mathbb{Z}) \subset \Gamma_{2,\infty}$, defined in (5.23), and its Fourier coefficients can be written after separating the finite contribution $C^F$, independent of $\Omega_2$, from the polar ones

$$
\int_{[0,1[^3} d^3\Omega_1 \frac{e^{i\pi(Q_1,Q_2)\begin{pmatrix} \rho & v \\ v & \sigma \end{pmatrix}\begin{pmatrix} Q_1 \\ Q_2 \end{pmatrix}}}{\Phi_{k-2}(\Omega)} = C^F_{k-2}(Q_1^2,Q_2^2,Q_1\cdot Q_2)
$$
$$
+ \sum_{\gamma \in GL(2,\mathbb{Z})/\mathrm{Dih}_4} c_k(-\tfrac{(sQ_1-qQ_2)^2}{2})c_k(-\tfrac{(pQ_2-rQ_1)^2}{2})\Big[ -\frac{\delta(\mathrm{tr}(\begin{pmatrix} 0 & 1/2 \\ 1/2 & 0 \end{pmatrix}\gamma^{\mathsf{T}}\Omega_2\gamma))}{4\pi}
$$
$$
+ \tfrac{(sQ_1-qQ_2)\cdot(pQ_2-rQ_1)}{2}(\mathrm{sign}((sQ_1-qQ_2)\cdot(pQ_2-rQ_1)) - \mathrm{sign}(\mathrm{tr}(\begin{pmatrix} 0 & 1/2 \\ 1/2 & 0 \end{pmatrix}\gamma^{\mathsf{T}}\Omega_2\gamma)))\Big],
\tag{5.57}
$$

where $\gamma = \begin{pmatrix} p & q \\ r & s \end{pmatrix}$, and the finite contributions $C^F_{k-2}(Q_1^2,Q_2^2,Q_1\cdot Q_2)$ are also invariant under $\begin{pmatrix} \widetilde{Q}_1 \\ \widetilde{Q}_2 \end{pmatrix} \to \gamma^{-1}\begin{pmatrix} \widetilde{Q}_1 \\ \widetilde{Q}_2 \end{pmatrix}$. The contributions of $1/\widetilde{\Phi}_{k-2}$ can be written similarly as

$$
\frac{1}{N}\int_{[0,N[\times[0,1[^2} d^3\Omega_1 \frac{e^{i\pi(Q_1,Q_2)\begin{pmatrix} \rho & v \\ v & \sigma \end{pmatrix}\begin{pmatrix} Q_1 \\ Q_2 \end{pmatrix}}}{\widetilde{\Phi}_{k-2}(\Omega)} = \widetilde{C}^F_{k-2}(Q_1^2,Q_2^2,Q_1\cdot Q_2)
$$
$$
+ \sum_{\gamma \in \Gamma_0(N)/\mathbb{Z}_2} c_k(-\tfrac{N(sQ_1-qQ_2)^2}{2})c_k(-\tfrac{(pQ_2-rQ_1)^2}{2})\Big[ -\frac{\delta(\mathrm{tr}(\begin{pmatrix} 0 & 1/2 \\ 1/2 & 0 \end{pmatrix}\gamma^{\mathsf{T}}\Omega_2\gamma))}{4\pi}
$$
$$
+ \tfrac{(sQ_1-qQ_2)\cdot(pQ_2-rQ_1)}{2}(\mathrm{sign}((sQ_1-qQ_2)\cdot(pQ_2-rQ_1)) - \mathrm{sign}(\mathrm{tr}(\begin{pmatrix} 0 & 1/2 \\ 1/2 & 0 \end{pmatrix}\gamma^{\mathsf{T}}\Omega_2\gamma)))\Big],
\tag{5.58}
$$

where $\mathbb{Z}_2 \ltimes \Gamma_0(N)$, the symmetry at the cusp, is equivalent to $GL(2,\mathbb{Z}) \cap M_{2,0}(N)$, and the stabilizer of $\begin{pmatrix} 0 & 1/2 \\ 1/2 & 0 \end{pmatrix}$ inside it is reduced to $\{\begin{pmatrix} 1 & 0 \\ 0 & 1 \end{pmatrix}, \begin{pmatrix} -1 & 0 \\ 0 & -1 \end{pmatrix}, \begin{pmatrix} 1 & 0 \\ 0 & -1 \end{pmatrix}\}$, leading the sum over $\Gamma_0(N)/\mathbb{Z}_2$.

1. The contributions from $(\widetilde{Q}_1,\widetilde{Q}_2) = (0,0)$ come in two classes: the ones associated to the zero mode $C^F_{k-2}(0,0,0) = \frac{48N}{N^2-1}$ and $\widetilde{C}^F_{k-2}(0,0,0) = -\frac{48}{N^2-1}$ (see (A.49) and (A.50)) that were absent for $N = 1$, and the ones coming from the delta function contribution in (5.57) and (5.58). The zero mode contribution is proportional to

$$
\int_{\mathcal{P}_2} \frac{d^3\Omega_2}{|\Omega_2|^{\frac{10-q}{2}}}\Big(N \sum_{\substack{A \in \\ M_2(\mathbb{Z})/GL(2,\mathbb{Z})}}{}' - v \sum_{\substack{A \in \\ M_{2,0}(N)/(\mathbb{Z}_2\ltimes\Gamma_0(N))}}{}' + v^2 \sum_{\substack{A \in \\ M_{2,00}(N)/GL(2,\mathbb{Z})}}{}' \Big)e^{-\pi\mathrm{Tr}[R^2\Omega_2^{-1}A^{\mathsf{T}}MA]}
$$
$$
= R^{2q-14}\pi^{\frac{2q-13}{2}}\Gamma(\tfrac{7-q}{2})\Gamma(\tfrac{6-q}{2})\Big(N \sum_{\substack{A \in \\ M_2(\mathbb{Z})/GL(2,\mathbb{Z})}}{}' - v \sum_{\substack{A \in \\ M_{2,0}(N)/(\mathbb{Z}_2\ltimes\Gamma_0(N))}}{}' + v^2 \sum_{\substack{A \in \\ M_{2,00}(N)/GL(2,\mathbb{Z})}}{}' \Big)\det A^{q-7}
$$
$$
= R^{2q-14}\xi(7-q)\xi(6-q)\big(N - v(1+N^{q-6}) + v^2 N^{q-7}\big),
\tag{5.59}
$$

where the integral is a matrix-variate Gamma integral [70] and the sums reduce to zeta functions using explicit representatives as (5.51).[16]

---

[16]Alternatively, the integral can be reduced to a beta integral over $r \in [0,1]$ using the substitution $v = \sqrt{\rho\sigma}r$.

With the same computation as in the preceding section, one obtains

$$
\begin{aligned}
G_{\alpha\beta,\gamma\delta}^{(p,q),2\mathrm{Ab}} &= -R^{2q-12}\frac{3c_k(0)^2}{64\pi^3}\big(\mathcal{E}^\star(\tfrac{8-q}{2},S)+\upsilon N^{\frac{q-8}{2}}\mathcal{E}^\star(\tfrac{8-q}{2},NS)\big)^2\delta_{\langle\alpha\beta},\delta_{\gamma\delta\rangle}\\
&\quad +\frac{18R^{2q-10}}{\pi^2}\xi(7-q)\xi(6-q)\frac{(N-\upsilon)(1-\upsilon N^{q-7})}{N^2-1}\delta_{\langle\alpha\beta},\delta_{\gamma\delta\rangle}\,,\\
G_{\alpha\beta,\rho\sigma}^{(p,q),2\mathrm{Ab}} &= -R^{2q-12}\frac{c_k(0)^2}{32\pi^3}\big(\mathcal{E}^\star(\tfrac{8-q}{2},S)+\upsilon N^{\frac{q-8}{2}}\mathcal{E}^\star(\tfrac{8-q}{2},NS)\big)\\
&\quad\times\Big[\tfrac{8-q}{2}\delta_{\rho\sigma}-2\mathcal{D}_{\rho\sigma}\Big]\tfrac{1}{2}\big(\mathcal{E}^\star(\tfrac{8-q}{2},S)+\upsilon N^{\frac{q-8}{2}}\mathcal{E}^\star(\tfrac{8-q}{2},NS)\big)\delta_{\alpha\beta}\\
&\quad +\frac{6(7-q)R^{2q-10}}{\pi^2}\xi(7-q)\xi(6-q)\frac{(N-\upsilon)(1-\upsilon N^{q-7})}{N^2-1}\delta_{\alpha\beta}\delta_{\rho\sigma}\,,\\
G_{\mu\nu,\rho\sigma}^{(p,q),2\mathrm{Ab}} &= -R^{2q-12}\frac{3c_k(0)^2}{64\pi^3}\Big[\tfrac{8-q}{2}\delta_{\langle\mu\nu},-2\mathcal{D}_{\langle\mu\nu},\Big]\big(\mathcal{E}^\star(\tfrac{8-q}{2},S)+\upsilon N^{\frac{q-8}{2}}\mathcal{E}^\star(\tfrac{8-q}{2},NS)\big)\\
&\quad\times\Big[\tfrac{8-q}{2}\delta_{\rho\sigma\rangle}-2\mathcal{D}_{\rho\sigma\rangle}\Big]\big(\mathcal{E}^\star(\tfrac{8-q}{2},S)+\upsilon N^{\frac{q-8}{2}}\mathcal{E}^\star(\tfrac{8-q}{2},NS)\big)\\
&\quad +\frac{9(6-q)(7-q)R^{2q-10}}{\pi^2}\xi(7-q)\xi(6-q)\frac{(N-\upsilon)(1-\upsilon N^{q-7})}{N^2-1}\delta_{\langle\mu\nu},\delta_{\rho\sigma\rangle}\,.
\end{aligned}
$$
(5.60)

Recall that $c_k(0)=\frac{24}{N+1}=k$ is the zero mode of $1/\Delta_k=\sum_m c_k(m)q^m$, and that $\delta_{\langle\alpha\beta},\delta_{\gamma\delta\rangle}=\frac{2}{3}(\delta_{\alpha\beta}\delta_{\gamma\delta}-\delta_{\alpha(\gamma}\delta_{\delta)\beta})$. As in the maximal rank case (5.26), the leading constant term in

$$
\mathcal{E}^\star(\tfrac{8-q}{2},S)\sim\xi(q-6)S_2^{\frac{q-6}{2}}+\xi(8-q)S_2^{\frac{8-q}{2}}
$$
(5.61)

reproduces the missing constant term in (4.60).

2. Contributions of non-zero vectors $(\widetilde{Q}_1,\widetilde{Q}_2)\in\Lambda_{p-2,q-2}^{\oplus 2}$ lead to the exponentially suppressed contributions written in (5.33). The measure of each Fourier mode will fall in three category, depending on the support of $(Q,P)$. The simplest one is for the most generic vector $Q\underline{\in}\Lambda_m^*$, $P\underline{\in}\Lambda_m$ – where we denote $X\underline{\in}\Lambda$ the strict inclusion of the vector $X$ in $\Lambda$, meaning that $X\in\Lambda$, $X\notin\Lambda[N]$ – for which only the first orbit in (5.51) of the second term in (5.24) contributes

$$
\upsilon\sum_{\substack{A=\left(\begin{smallmatrix}k&j\\0&p\end{smallmatrix}\right),\,0\le j<k\\p\ne 0\\A^{-1}\left(\begin{smallmatrix}Q\\P\end{smallmatrix}\right)\in\Lambda_m^*\oplus\Lambda_m}}|A|^{q-7}\widetilde{C}_{k-2}\Big[A^{-1}\big(\begin{smallmatrix}-|Q|^2&-Q\cdot P\\-Q\cdot P&-|P|^2\end{smallmatrix}\big)A^{-\mathsf{T}};A^{\mathsf{T}}\Omega_2^\star A\Big],
$$
(5.62)

where the $N$ factor comes from the width of the integration domain $(\mathbb{R}/N\mathbb{Z})$.

For less generic vectors $Q\underline{\in}\Lambda_m^*$, $P\underline{\in}N\Lambda_m^*$, one must add to (5.62) the second orbit of (5.51), allowing to rewrite the two as a sum over $M_{2,0}(N)/(\mathbb{Z}_2\ltimes\Gamma_0(N))$ defined in (5.51), as well as the contribution from the last term of (5.24). We obtain

$$
\begin{aligned}
&\upsilon\sum_{\substack{A\in M_{2,0}(N)/[\mathbb{Z}_2\ltimes\Gamma_0(N)]\\A^{-1}\left(\begin{smallmatrix}Q\\P\end{smallmatrix}\right)\in\Lambda_m^*\oplus\Lambda_m}}|A|^{q-7}\widetilde{C}_{k-2}\Big[A^{-1}\big(\begin{smallmatrix}-|Q|^2&-Q\cdot P\\-Q\cdot P&-|P|^2\end{smallmatrix}\big)A^{-\mathsf{T}};A^{\mathsf{T}}\Omega_2^\star A\Big]\\
&+\upsilon^2\sum_{\substack{A\in M_2(N)/GL(2,\mathbb{Z})\\A^{-1}\left(\begin{smallmatrix}Q\\P/N\end{smallmatrix}\right)\in\Lambda_m^*\oplus\Lambda_m^*}}N^{q-8}|A|^{q-7}C_{k-2}\Big[A^{-1}\big(\begin{smallmatrix}-N|Q|^2&-Q\cdot P\\-Q\cdot P&-|P|^2/N\end{smallmatrix}\big)A^{-\mathsf{T}};A^{\mathsf{T}}\Omega_2^\star A\Big],
\end{aligned}
$$
(5.63)

where in the second line, $N$ factors come from the width of the integration domain $(\mathbb{R}/N\mathbb{Z})^3$, as well as the argument of $1/\Phi_{k-2}(N\Omega)$, and the magnetic vector is rescaled

$P \rightarrow P/N$, allowing us to use $M_2(N)/GL(2,\mathbb{Z})$ instead of $M_{2,00}(N)/GL(2,\mathbb{Z})$ (5.54) for simplicity.

Finally, for vectors $Q \in \Lambda_m$, $P \in \Lambda_m$, one must add to (5.62) the contribution from the first term of (5.24). One thus obtain the full measure as

$$
\begin{aligned}
\bar{C}_{k-2}(Q,P,\Omega^\star) =& \sum_{\substack{A \in M_2(\mathbb{Z})/GL(2,\mathbb{Z}) \\ A^{-1}\binom{Q}{P} \in \Lambda_m \oplus \Lambda_m}} |A|^{q-7} C_{k-2}\left[A^{-1}\begin{pmatrix} -|Q|^2 & -Q\cdot P \\ -Q\cdot P & -|P|^2 \end{pmatrix}A^{-\intercal}; A^\intercal \Omega_2^\star A\right] \\
&+ \upsilon \sum_{\substack{A \in M_{2,0}(N)/[\mathbb{Z}_2 \ltimes \Gamma_0(N)] \\ A^{-1}\binom{Q}{P} \in \Lambda_m^* \oplus \Lambda_m}} |A|^{q-7} \widetilde{C}_{k-2}\left[A^{-1}\begin{pmatrix} -|Q|^2 & -Q\cdot P \\ -Q\cdot P & -|P|^2 \end{pmatrix}A^{-\intercal}; A^\intercal \Omega_2^\star A\right] \\
&+ \upsilon^2 \sum_{\substack{A \in M_2(\mathbb{Z})/GL(2,\mathbb{Z}) \\ A^{-1}\binom{Q}{P/N} \in \Lambda_m^* \oplus \Lambda_m^*}} N^{q-8}|A|^{q-7} C_{k-2}\left[A^{-1}\begin{pmatrix} -N|Q|^2 & -Q\cdot P \\ -Q\cdot P & -|P|^2/N \end{pmatrix}A^{-\intercal}; A^\intercal \Omega_2^\star A\right].
\end{aligned}
\tag{5.64}
$$

Finally, there are also contributions from rank two non-abelian orbits where the two rows $(\mathbf{n}_1, \mathbf{m}_1)$ and $(\mathbf{n}_2, \mathbf{m}_2)$ have non vanishing symplectic product $\mathbf{n}_1 \cdot \mathbf{m}_2 - \mathbf{m}_1 \cdot \mathbf{n}_2 \neq 0$, but as mentioned in the previous subsection, it is more convenient to obtain them from the Fourier coefficients in the degeneration $(p,q) \rightarrow (p-1, q-1)$, as explained in Appendix G.

### 5.3  Large radius limit and BPS dyon counting

We now apply the results in §5.1 and 5.2 for $(p,q) = (2k, 8)$ and $\Lambda_{p-2,q-2} = \Lambda_m$, to discuss the limit of the exact $\nabla^2(\nabla\phi)^4$ couplings in three-dimensional CHL orbifolds, in the limit where one circle inside $T^7$ (orthogonal to the circle involved in the orbifold action) decompactifies. We regularize the coupling coefficient by analytic coninuation of $q = 8 + 2\epsilon$, and we substract the pole at $\epsilon = 0$. We find that the conjectured exact $\nabla^2(\nabla\phi)^4$ coupling (1.7) has the large radius expansion

$$
G^{(2k,8)}_{\alpha\beta,\gamma\delta} = G^{(0)}_{\alpha\beta,\gamma\delta} + G^{(1)}_{\alpha\beta,\gamma\delta} + G^{(2)}_{\alpha\beta,\gamma\delta} + G^{(TN)}_{\alpha\beta,\gamma\delta}
\tag{5.65}
$$

corresponding to the constant term, 1/2-BPS and 1/4-BPS Abelian Fourier modes and finally, the non-Abelian Fourier modes with non-zero Taub-NUT charge discussed in Appendix G.

#### 5.3.1  Effective action in $D = 4$

The constant term in (5.65) takes the form

$$
G^{(0)}_{\alpha\beta,\gamma\delta} = R^4 G^{(D=4)}_{\alpha\beta,\gamma\delta} + \frac{\zeta(3)}{8\pi}(k-12)R^6 \delta_{\langle\alpha\beta}, \delta_{\gamma\delta\rangle} + \mathcal{O}(e^{-2\pi R}).
\tag{5.66}
$$

The first term originates from orbits of rank 0 (5.36), rank-1 (5.44) and Abelian rank-2 (5.60), and combines all terms proportional to $R^4$ that survive in the decompactification limit. The second term comes from (5.60), and can be ascribed to the 2-loop sunset diagram shown in Figure 1 c), with Kaluza–Klein states running in the loops. Its coefficient vanishes in the maximal rank case. The exponentially suppressed contributions of order $e^{-R}$ and $e^{-R^2}$ are missed by the unfolding procedure, but they must be present because of the differential equation (2.26). We shall return to them in the next subsection.

If our Ansatz (1.7) for the exact $\nabla^2(\nabla\phi)^4$ couplings in $D = 3$ is correct, the term proportional to $R^4$ in (5.66) must reproduce the exact $\nabla^2 F^4$ couplings in four dimensions, up to

logarithmic corrections in $R$ due to the mixing between local and non-local couplings in $D = 4$. For the maximal rank case, we find

$$G^{(D=4)}_{\alpha\beta,\gamma\delta}(S,\varphi) = \widehat{G}^{(24,6)}_{\alpha\beta,\gamma\delta}(\varphi) - \frac{3}{4\pi}\delta_{\langle\alpha\beta}\delta_{\gamma\delta\rangle}\Big(\hat{\mathcal{E}}_1(S) + \frac{3}{\pi}\log R\Big)^2 - \frac{1}{4}\delta_{\langle\alpha\beta,}\Big(\hat{\mathcal{E}}_1(S) + \frac{3}{\pi}\log R\Big)\widehat{G}^{(24,6)}_{\gamma\delta\rangle}(\varphi)\,,$$
(5.67)

where we used the definition (5.11)

$$\mathcal{E}_s(S) = \frac{1}{\xi(2s)}\mathcal{E}^{\star}(s,S) = S_2^s + \frac{\xi(2s-1)}{\xi(2s)}S_2^{1-s} + \mathcal{O}(e^{-2\pi S_2})\,,$$
(5.68)

and the regularized value at $s = 1$,

$$\hat{\mathcal{E}}_1(S) = \lim_{s\to 1}\left[\mathcal{E}_s(S) - \frac{3(\frac{A_G^{12}}{4\pi})^{2(s-1)}}{\pi(s-1)}\right] = -\frac{1}{4\pi}\log(S_2^{12}|\Delta(S)|)\,,$$
(5.69)

where $A_G = e^{\frac{1}{12} - \zeta'(-1)}$ is the Glaisher-Kinkelin constant.

Recalling that $S_2 = 1/g_4^2$, we see that the first term in (5.67) indeed reproduces the two-loop contribution to the $\nabla^2 F^4$ coupling in $D = 4$, while the two other terms reproduce the tree-level and one-loop contributions to the same coupling, along with non-perturbative NS5-brane corrections of order $e^{-2\pi S_2}$. Because there is no holomorphic modular form of weight zero for $SL(2,\mathbb{Z})$, supersymmetry Ward identities and U-duality determine uniquely this non-perturbative coupling from its perturbative expansion.

For the CHL orbifolds with $N = 2,3,5,7$, we find instead

$$\begin{aligned} G^{(D=4)}_{\alpha\beta,\gamma\delta}(S,\varphi) = \;& \widehat{G}^{(2k-2,6)}_{\alpha\beta,\gamma\delta}(\varphi) - \frac{3}{4\pi}\delta_{\langle\alpha\beta}\delta_{\gamma\delta\rangle}\Big(\frac{\hat{\mathcal{E}}_1(NS) + \hat{\mathcal{E}}_1(S) + \frac{6}{\pi}\log R}{N+1}\Big)^2 \\ & - \frac{1}{4(N+1)}\delta_{\langle\alpha\beta,}\Big(\Big(\frac{N\hat{\mathcal{E}}_1(NS) - \hat{\mathcal{E}}_1(S)}{N-1} + \frac{6}{\pi}\log R\Big)\widehat{G}^{(2k-2,6)}_{\gamma\delta\rangle}(\varphi) \\ & + \Big(\frac{N\hat{\mathcal{E}}_1(S) - \hat{\mathcal{E}}_1(NS)}{N-1} + \frac{6}{\pi}\log R\Big)^\varsigma\widehat{G}^{(2k-2,6)}_{\gamma\delta\rangle}(\varphi)\Big)\,, \end{aligned}$$
(5.70)

which is manifestly invariant under the Fricke duality $S \mapsto -1/(NS)$, $\varphi \to \varsigma \cdot \varphi$ [27]. In the weak coupling limit $S_2 \to +\infty$, this again reproduces the tree-level, one-loop and two-loop contributions to the $\nabla^2 F^4$ coupling in $D = 4$ (discarding the log terms)

$$G^{(D=4)}_{\alpha\beta,\gamma\delta}(S,\varphi) = \;\widehat{G}^{(2k-2,6)}_{\alpha\beta,\gamma\delta}(\varphi) - \frac{3}{4\pi}\delta_{\langle\alpha\beta,}\delta_{\gamma\delta\rangle}S_2{}^2 - \frac{1}{4}\delta_{\langle\alpha\beta,}\widehat{G}^{(2k-2,6)}_{\gamma\delta\rangle}(\varphi)S_2 + \mathcal{O}(e^{-2\pi S_2})\,.$$

This agreement is of course guaranteed by the similar agreement in $D = 3$ discussed in §4.3. Since there are no cuspidal forms of weight zero for $\Gamma_0(N)$, (5.70) is in fact the unique non-perturbative completion of the perturbative coupling consistant with supersymmetry Ward identities and U-duality, including Fricke duality.[17]

Other tensorial components $G_{\alpha\beta,\mu\nu}$ correspond instead to $\mathcal{R}^2 F^2$ couplings in $D = 4$, which we refrain from discussing in detail.

---

[17]The square of $\frac{\hat{\mathcal{E}}_1(NS) + \hat{\mathcal{E}}_1(S)}{N+1}$ is determined by supersymmetry. The combination $\frac{\hat{\mathcal{E}}_1(NS) + \hat{\mathcal{E}}_1(S)}{N+1}(\widehat{G}^{(2k-2,6)}_{\alpha\beta}(\varphi) + {}^\varsigma\widehat{G}^{(2k-2,6)}_{\alpha\beta}(\varphi)) = \frac{\hat{\mathcal{E}}_1(NS) + \hat{\mathcal{E}}_1(S)}{2}F^{(2k-2,6)\gamma}_{\alpha\beta\gamma}(\varphi)$ is determined with a fixed coefficient by the source term in the differential equation enforced by supersymmetry whereas the coefficient of $\frac{\hat{\mathcal{E}}_1(NS) - \hat{\mathcal{E}}_1(S)}{N-1}(\widehat{G}^{(2k-2,6)}_{\alpha\beta}(\varphi) - {}^\varsigma\widehat{G}^{(2k-2,6)}_{\alpha\beta}(\varphi))$ is determined by matching the perturbative expansion.

### 5.3.2 Contributions from 1/4-BPS instantons

Exponentially suppressed corrections arise from the rank one orbits (5.22), the Abelian rank two orbits (5.33), and the non-Abelian rank two (G.9). In this section, we focus on the contributions from the the Abelian rank two orbits, which provide the Abelian Fourier coefficients for generic 1/4-BPS charges.[18] These Fourier coefficients can be interpreted as non-perturbative corrections associated to space-time instantons corresponding to 1/4-BPS black holes wrapping the Euclidean time circle.

Decomposing

$$
G^{(2)}_{ab,cd} = \sum_{\substack{\Gamma \in \Lambda_m^* \oplus \Lambda_m \\ Q \wedge P \neq 0}} G^{(2,\Gamma)}_{ab,cd}\, e^{2\pi i(a_1 Q + a_2 P)}, \tag{5.71}
$$

with $\Gamma = (Q, P)$, using (5.24) and the change of variable $\Omega_2 \to A^\intercal \Omega_2 A$, one obtains

$$
G^{(2,\Gamma)}_{ab,cd} = 2R^4 \int_{\mathcal{P}} d^3\Omega_2\, \bar{C}_{k-2}(Q,P;\Omega_2)\, P_{ab,cd}(Q_L,P_L,\Omega_2)\, e^{-\pi \mathrm{Tr}\left[\frac{R^2}{S_2}\Omega_2^{-1}\left(\begin{smallmatrix} 1 & S_1 \\ S_1 & |S|^2 \end{smallmatrix}\right) + 2\Omega_2 \left(\begin{smallmatrix} Q_R^2 & Q_R P_R \\ Q_R P_R & P_R^2 \end{smallmatrix}\right)\right]}, \tag{5.72}
$$

with [19]

$$
P_{ab,cd}(Q_L,P_L,\Omega_2) = \left(\mathcal{P}_{ab,cd}\left(\tfrac{\partial}{\partial y}\right) e^{\pi\sqrt{2}Ry_{i\mu}(\Omega_2^{-1})^{ij}v^{\intercal j\mu} + 2\pi i y_{i\alpha}\Gamma_L^{i\alpha} - \frac{\pi}{2}y_{i\alpha}(\Omega_2^{-1})^{ij}y_j{}^{\alpha}}\right)\Big|_{y=0}. \tag{5.73}
$$

The summation measure $\bar{C}(Q,P,\Omega_2)$ depends both on the charge $\Gamma = (Q,P)$ and on $\Omega_2 \in \mathcal{P}$, and is given for the maximal rank model by (cf. (5.34))

$$
\bar{C}(Q,P;\Omega_2) = \sum_{\substack{A \in M_2(\mathbb{Z})/GL(2,\mathbb{Z}) \\ A^{-1}\Gamma \in \Lambda_{22,6}\oplus\Lambda_{22,6}}} |A|\, C\left[A^{-1}\left(\begin{smallmatrix} -Q^2 & -Q\cdot P \\ -Q\cdot P & -P^2 \end{smallmatrix}\right)A^{-\intercal}; A^\intercal\Omega_2 A\right], \tag{5.74}
$$

where $\Lambda_{22,6} = \Lambda_m$ is the magnetic lattice of the full rank model, and $C\left[\left(\begin{smallmatrix} 2m & l \\ l & 2n \end{smallmatrix}\right);\Omega_2\right]$ are the Fourier coefficients of $1/\Phi_{10}$ defined in (5.25). For CHL models with $N = 2,3,5,7$, it is instead given by (cf. (5.64))

$$
\begin{aligned}
\bar{C}_{k-2}(Q,P;\Omega_2) = &\sum_{\substack{A \in M_2(\mathbb{Z})/GL(2,\mathbb{Z}) \\ A^{-1}\left(\begin{smallmatrix} Q \\ P \end{smallmatrix}\right)\in\Lambda_m\oplus\Lambda_m}} |A| C_{k-2}\left[A^{-1}\left(\begin{smallmatrix} -Q^2 & -Q\cdot P \\ -Q\cdot P & -P^2 \end{smallmatrix}\right)A^{-\intercal}; A^\intercal\Omega_2 A\right] \\
&+ \sum_{\substack{A \in M_{2,0}(N)/[\mathbb{Z}_2\ltimes\Gamma_0(N)] \\ A^{-1}\left(\begin{smallmatrix} Q \\ P \end{smallmatrix}\right)\in\Lambda_m^*\oplus\Lambda_m}} |A| \widetilde{C}_{k-2}\left[A^{-1}\left(\begin{smallmatrix} -Q^2 & -Q\cdot P \\ -Q\cdot P & -P^2 \end{smallmatrix}\right)A^{-\intercal}; A^\intercal\Omega_2 A\right] \\
&+ \sum_{\substack{A \in M_2(\mathbb{Z})/GL(2,\mathbb{Z}) \\ A^{-1}\left(\begin{smallmatrix} Q \\ P/N \end{smallmatrix}\right)\in\Lambda_m^*\oplus\Lambda_m^*}} |A| C_{k-2}\left[A^{-1}\left(\begin{smallmatrix} -NQ^2 & -Q\cdot P \\ -Q\cdot P & -P^2/N \end{smallmatrix}\right)A^{-\intercal}; A^\intercal\Omega_2 A\right], \tag{5.75}
\end{aligned}
$$

where $C_{k-2}\left[\left(\begin{smallmatrix} 2m & l \\ l & 2n \end{smallmatrix}\right);\Omega_2\right]$ and $\widetilde{C}_{k-2}\left[\left(\begin{smallmatrix} 2m & l \\ l & 2n \end{smallmatrix}\right);\Omega_2\right]$ denote the Fourier coefficients of $1/\Phi_{k-2}(\Omega)$ and $1/\widetilde{\Phi}_{k-2}(\Omega)$ given in (5.57), (5.58).

As emphasized earlier, $1/\Phi_{k-2}(\Omega)$ and $1/\widetilde{\Phi}_{k-2}(\Omega)$ are meromorphic functions with poles, so that their Fourier coefficients are piecewise constant functions of $\Omega_2$, with discontinuities as well as delta-function singularities at the boundary between distinct chambers (moreover, they

---

[18]The dimension of the set of generic 1/4-BPS charges, plus one for the Taub-NUT charge, is equal to the Kostant–Kirillov dimension of the automorphic representation attached to $G_{ab,cd}$, see the end of section 3.1.

[19]When $\alpha\beta\gamma\delta$ lie along the $O(2k-2,6)$ directions, $P_{\alpha\beta,\gamma\delta}(Q_L,P_L,\Omega_2)$ reduces to the polynomial in (2.31).

are strictly speaking well-defined only for $|\Omega_2| > \frac{1}{4}$, since the contour $\mathcal{C} = [0,1]^3$ generically crosses the poles for lower values of $|\Omega_2|$). Due to this non-trivial $\Omega_2$-dependence, one cannot compute the integral (5.72) analytically, but one may analyze its asymptotic expansion at large radius.

For generic moduli $S$ and $\varphi$, the integral is dominated by a saddle point at $\Omega_2 = \Omega_2^\star$ (5.31), in the neighborhood of which the Fourier coefficients of $1/\Phi_{k-2}(\Omega)$ and $1/\widetilde{\Phi}_{k-2}(\Omega)$ are constant. One can compute the leading contribution in the saddle point approximation by integrating (5.72) with $\bar{C}_{k-2}(Q,P;\Omega_2) \sim \bar{C}_{k-2}(Q,P;\Omega_2^\star)$ kept constant in the integrand. Using (5.33) and the identities [13, (20)]

$$
\begin{aligned}
\widetilde{B}_{3/2}(Z) &= \frac{\pi K_0\big(2\pi\mathcal{M}(Z)\big)}{\det(Z)^{1/4}} + \frac{\pi\mathcal{M}(Z)K_1\big(2\pi\mathcal{M}(Z)\big)}{2\det(Z)^{3/2}} \,, \\
\widetilde{B}_{1/2}(Z) &= \frac{\pi K_0\big(2\pi\mathcal{M}(Z)\big)}{\det(Z)^{1/4}} \,,
\end{aligned}
\tag{5.76}
$$

where $\mathcal{M}(Z) = \sqrt{2\sqrt{\det Z} + \mathrm{tr}(Z)}$ (such that $\mathcal{M}(2R^2 v\left(\begin{smallmatrix} Q_R^2 & Q_R P_R \\ Q_R P_R & P_R^2 \end{smallmatrix}\right)v^{\mathsf{T}}) = R\mathcal{M}(\Gamma)$), the resulting 1/4-BPS Abelian Fourier coefficients in this approximation can be expressed in terms of the standard modified Bessel functions,

$$
\begin{aligned}
G_{\alpha\beta,\gamma\delta}^{(2,\Gamma)} &\sim \frac{9}{16}R^5\,\bar{C}_{k-2}(Q,P;\Omega_2^\star) \\
&\times \Bigg( \frac{2\pi}{R^2}\frac{Q_{L\langle\alpha}Q_{L\beta},P_{L\gamma}P_{L\delta\rangle}}{|Q_R \wedge P_R|^2}\Bigg[K_0(2\pi R\mathcal{M}(\Gamma)) + \frac{R\mathcal{M}(\Gamma)}{4R^2|2Q_R \wedge P_R|}K_1(2\pi R\mathcal{M}(\Gamma))\Bigg] \\
&+ \frac{1}{\pi}\delta_{\langle\alpha\beta,}\frac{\Gamma_{L\gamma}{}^{\kappa}\Gamma_{L\delta\rangle}{}^{\lambda}}{|Q_R \wedge P_R|}\frac{\partial}{\partial Z^{\kappa\lambda}}\Big[2\sqrt{|Z|}K_0(2\pi\mathcal{M}(Z)) + \mathcal{M}(Z)K_1(2\pi\mathcal{M}(Z))\Big]\Bigg|_{Z=2R^2 v\left(\begin{smallmatrix} Q_R^2 & Q_R P_R \\ Q_R P_R & P_R^2 \end{smallmatrix}\right)v^{\mathsf{T}}} \\
&+ \frac{1}{4\pi|Q_R \wedge P_R|}\delta_{\langle\alpha\beta,}\delta_{\gamma\delta\rangle}K_0(2\pi R\mathcal{M}(\Gamma))\Bigg]\Bigg) \,,
\end{aligned}
\tag{5.77}
$$

where $\Gamma_{L\gamma}{}^{\kappa} = \frac{1}{\sqrt{S_2}}(Q_{L\gamma}+S_1 P_{L\gamma}, S_2 P_{L\gamma})$. This leading contribution can be ascribed to instantons of charge $\Gamma$ associated to 1/4-BPS black holes (including bound states of two 1/2-BPS black holes) wrapping the Euclidean time circle. It is indeed exponentially suppressed in $e^{-2\pi R\mathcal{M}(\Gamma)}$ for $\mathcal{M}(\Gamma)$ (2.5) the BPS mass of a black hole of charge $\Gamma$, and it is weighted by the measure factor $\bar{C}_{k-2}(Q,P;\Omega_2^\star)$. For a primitive charge $\Gamma$, i.e. such that there is no $d \neq 1$ with $d^{-1}\Gamma \in \Lambda_m^\star \oplus \Lambda_m$, the only matrix $A$ contributing to the measure is $A = 1$ and one can interpret the measure factor (up to an overall sign) as the helicity supertrace counting string theory states of charge $\Gamma$, as advocated in the introduction (2.14),

$$
\bar{C}_{k-2}(Q,P;\Omega_2^\star) = (-1)^{Q\cdot P+1}\Omega_6(Q,P,S,\varphi) \,.
\tag{5.78}
$$

The value of $\Omega_2$ at the saddle point (5.31) reproduces the contour prescription of [9,33] when both electric and magnetic charges are separately primitive in $\Lambda_m^\star$ and $\Lambda_m$ and $d^{-1}Q \wedge P \in \Lambda_m^\star \wedge \Lambda_m$ for $d = 1$ only. More generally, the contour prescription depends on the set of matrices $A$ dividing $(Q,P)$ in the electromagnetic lattice. For example in the maximal rank case, all primitive charges $(Q,P)$ are in the U-duality orbit of a charge of the form [71]

$$
Q = e_1 + q\,e_2 \,, \qquad P = p\,e_2 \,, \qquad Q \wedge P = p\,e_1 \wedge e_2 \,,
\tag{5.79}
$$

with $e_1$ and $e_2$ primitive in $\Lambda_{22,6}$. The integer $p$ is sometimes known as the 'torsion'. In that case (5.74) simplifies to

$$
\bar{C}(Q,P;\Omega_2^\star) = \sum_{\substack{d\geq 1 \\ d|p}} d\,C\Big[\Big(\begin{smallmatrix} Q^2 & QP/d \\ QP/d & P^2/d^2 \end{smallmatrix}\Big),\Big(\begin{smallmatrix} 1 & 0 \\ 0 & d \end{smallmatrix}\Big)\Omega_2^\star\Big(\begin{smallmatrix} 1 & 0 \\ 0 & d \end{smallmatrix}\Big)\Big] \,,
\tag{5.80}
$$

in agreement with the prescription in [41,72], with additional fineprint on the contour of integration. If we consider the same charge configuration (5.79) in CHL orbifolds for $e_1$ primitive in $\Lambda_m^*$ and not in $\Lambda_m$, $e_2$ primitive in $\Lambda_m$ and not in $N\Lambda_m^*$, and with $p$ not divisible by $N$, such that it corresponds to a twisted state, only the second line in (5.75) contributes and the result reduces similarly to

$$\bar{C}_{k-2}(Q,P;\Omega_2^\star) = \sum_{\substack{d \geq 1 \\ d | p}} d\ \widetilde{C}_{k-2}\left[ \left( \begin{smallmatrix} Q^2 & QP/d \\ QP/d & P^2/d^2 \end{smallmatrix} \right), \left( \begin{smallmatrix} 1 & 0 \\ 0 & d \end{smallmatrix} \right) \Omega_2^\star \left( \begin{smallmatrix} 1 & 0 \\ 0 & d \end{smallmatrix} \right) \right], \tag{5.81}$$

in agreement with [6] for $p = 1$. For general primitive charges such that $Q$ can be in $\Lambda_m$ and $P$ in $N\Lambda_m^*$, all three terms contribute to the helicity supertrace, and the result is manifestly invariant under U-duality including Fricke duality.

### 5.3.3 Contributions from pairs of 1/2-BPS instantons

Let us now discuss corrections to the saddle point approximation to (5.72). In Appendix F we estimate the contributions to $G_{\alpha\beta,\gamma\delta}^{(2,\Gamma)}$ due to the deviation of $\bar{C}_{k-2}(Q,P,\Omega_2)$ from its saddle point value $\bar{C}_{k-2}(Q,P,\Omega_2^*)$. In the range[20] $|\Omega_2| > \frac{1}{4}$, the deviation is due to the poles occuring when $n_1\sigma_2 - m^1\rho_2 + jv_2 = 0$ with $4n_1m^1 + j^2 = 0$, resulting in the discontinuities and delta-function singularities of $C_{k-2}(Q,P,\Omega_2^*)$ and $\widetilde{C}_{k-2}(Q,P,\Omega_2^*)$ on $\mathcal{P}$ shown in (5.25), (5.57) and (5.58). In Appendix F.1, we show that these contributions are exponentially suppressed in $e^{-2\pi R(\mathcal{M}(\Gamma_1)+\mathcal{M}(\Gamma_2))}$, and can therefore be ascribed to two-instanton effects associated to two unbounded 1/2-BPS states of charges $\Gamma_1$ and $\Gamma_2$.

For fixed total charge $\Gamma$, we expect contributions from all pairs of 1/2-BPS states with charges $\Gamma_1$ and $\Gamma_2$ such that $\Gamma = \Gamma_1 + \Gamma_2$. We show in Appendix C that a general such splitting is parametrized by a non-degenerate matrix $B = \left( \begin{smallmatrix} p & q \\ r & s \end{smallmatrix} \right) \in M_2(\mathbb{Z})$, such that

$$\binom{Q_1}{P_1} = \binom{p}{r}\frac{sQ-qP}{ps-qr} = B\pi_1 B^{-1}\binom{Q}{P}, \quad \binom{Q_2}{P_2} = \binom{q}{s}\frac{pP-rQ}{ps-qr} = B\pi_2 B^{-1}\binom{Q}{P}, \tag{5.82}$$

where $\pi_1 = \left( \begin{smallmatrix} 1 & 0 \\ 0 & 0 \end{smallmatrix} \right)$ and $\pi_2 = \left( \begin{smallmatrix} 0 & 0 \\ 0 & 1 \end{smallmatrix} \right)$. All splittings of a given charge $\Gamma$ are in one-to-one correspondence with the matrices $B \in M_2(\mathbb{Z})/\text{Stab}(\pi_i)$ such that $B\pi_1 B^{-1}\Gamma \in \Lambda_m^* \oplus \Lambda_m$ with

$$M_2(\mathbb{Z})/\text{Stab}(\pi_i) = \left\{ \gamma \cdot \begin{pmatrix} 1 & j' \\ 0 & k' \end{pmatrix}, \quad \gamma \in GL(2,\mathbb{Z})/\text{Dih}_4, \quad 0 \leq j' < k', \quad (j',k') = 1 \right\}. \tag{5.83}$$

In the following it prove convenient to use an equivalent unimodular representative

$$\hat{B} = B\begin{pmatrix} 1 & 0 \\ 0 & |B|^{-1} \end{pmatrix} = \gamma \cdot \begin{pmatrix} 1 & \frac{j'}{k'} \\ 0 & 1 \end{pmatrix}, \tag{5.84}$$

in $SL(2,\mathbb{Q})/\text{Stab}(\pi_i,\mathbb{Q})$, where $\text{Stab}(\pi_i,\mathbb{Q})$ is the stabilizer of the doublet $\pi_i$ in $SL(2,\mathbb{Q})$.

We show in Appendix C that the summation measure (5.74) on the domain $|\Omega_2| > \frac{1}{4}$ (taking into account the discontinuities displayed in (5.25)) reads (focusing on the maximal rank case

---

[20]In the range $|\Omega_2| < \frac{1}{4}$, there are additional contributions from 'deep poles' of the form (F.10) with $n_2 \neq 0$ which must be avoided in order to define the Fourier coefficient $\bar{C}(Q,P,\Omega_2)$. In Appendix (F.2), we show that irrespective of the detailed prescription for avoiding these poles, the contribution from the region $|\Omega_2| < \frac{1}{(4n_2^2)}$ is exponentially suppressed in $e^{-2\pi R^2|2n_2|}$, and can be ascribed to pairs of Taub-NUT instanton anti-instantons of charge $\pm n_2$.

for simplicity)

$$\bar{C}(Q,P;\Omega_2) = \sum_{\substack{A\in M_2(\mathbb{Z})/GL(2,\mathbb{Z}) \\ A^{-1}\Gamma\in\Lambda_m\oplus\Lambda_m}} |A| \, C^F\Big[A^{-1}\begin{pmatrix} -Q^2 & -Q\cdot P \\ -Q\cdot P & -P^2 \end{pmatrix}A^{-\intercal}\Big] \tag{5.85}$$

$$+\sum_{\substack{\Gamma_i\in\Lambda_m\oplus\Lambda_m \\ Q_i\wedge P_i=0,\,\Gamma_1+\Gamma_2=\Gamma}} \bar{c}(\Gamma_1)\bar{c}(\Gamma_2)\Big(-\frac{\delta([\hat{B}^\intercal\Omega_2\hat{B}]_{12})}{4\pi} + \frac{\langle\Gamma_1,\Gamma_2\rangle}{2}\big(\text{sign}(\langle\Gamma_1,\Gamma_2\rangle) - \text{sign}([\hat{B}^\intercal\Omega_2\hat{B}]_{12})\big)\Big),$$

with $\hat{B}\in SL(2,\mathbb{Q})/\text{Stab}(\pi_i,\mathbb{Q})$ determined such that $\Gamma_i = \hat{B}\pi_i\hat{B}^{-1}\Gamma$ and where $[\hat{B}^\intercal\Omega_2\hat{B}]_{ij}$ denotes the entrises $ij$ of the matrix.

To interpret the second line, recall that the central charge $Z = \frac{2}{\sqrt{S_2}}(Q_R + SP_R)$ for an arbitrary 1/4-BPS state decomposes into orthogonal components $Z = Z_+ + Z_-$ with

$$Z_\pm = \frac{1}{\sqrt{S_2}}\Big[(1,S)\cdot\begin{pmatrix} Q_R \\ P_R \end{pmatrix} \pm \frac{i}{|Q_R\wedge P_R|}(-S,1)\cdot\begin{pmatrix} P_R^2 Q_R - (Q_R\cdot P_R)P_R \\ Q_R^2 P_R - (Q_R\cdot P_R)Q_R \end{pmatrix}\Big]. \tag{5.86}$$

The BPS mass is $\mathcal{M}(Q,P) = |Z_+|$. It is convenient to write $Z_{+\hat{\alpha}} = (z_1 + iz_2)_{\hat{\alpha}}\mathcal{M}(Q,P)$ with $z_1$ and $z_2$ vectors of $SO(6)$ satisfying

$$z_1^2 + z_2^2 = 1\,, \quad z_1\cdot\frac{Q_R+S_1 P_R}{\sqrt{S_2}} + z_2\cdot\frac{S_2 P_R}{\sqrt{S_2}} = 2\mathcal{M}(Q,P)\,, \quad z_1\cdot\frac{S_2 P_R}{\sqrt{S_2}} - z_2\cdot\frac{Q_R+S_1 P_R}{\sqrt{S_2}} = 0\,. \tag{5.87}$$

The matrix $\Omega_2^\star$ at the saddle point determines precisely this decomposition through

$$\begin{pmatrix} z_1 \\ z_2 \end{pmatrix} = \frac{1}{\sqrt{S_2}}\begin{pmatrix} S_2 & 0 \\ -S_1 & 1 \end{pmatrix}\frac{1}{R}\Omega_2^\star\begin{pmatrix} Q_R \\ P_R \end{pmatrix}. \tag{5.88}$$

A generic two-center 1/4-BPS solution with total charge $(Q,P)$ is written in terms of the harmonic functions [21]

$$(\mathcal{H}^I,\mathcal{K}^I) = \frac{(Q_1^I,P_1^I)}{|x-x_1|} + \frac{(Q_2^I,P_2^I)}{|x-x_2|} - p_{R\hat{\alpha}}{}^I\frac{1}{\sqrt{S_2}}\begin{pmatrix} S_2 & -S_1 \\ 0 & 1 \end{pmatrix}\begin{pmatrix} z_1^{\hat{\alpha}} \\ z_2^{\hat{\alpha}} \end{pmatrix}, \tag{5.89}$$

and is regular away from the points $x_1$ and $x_2$ provided the distance $|x_1 - x_2|$ satisfies

$$\frac{\langle\Gamma_1,\Gamma_2\rangle}{|x_1 - x_2|} = -z_1\cdot\frac{S_2 P_{1R}}{\sqrt{S_2}} + z_2\cdot\frac{Q_{1R}+S_1 P_{1R}}{\sqrt{S_2}} = -\frac{|Q_R\wedge P_R|}{R}[\hat{B}^\intercal\Omega_2^\star\hat{B}]_{12}\,, \tag{5.90}$$

which requires that $[\hat{B}^\intercal\Omega_2^\star\hat{B}]_{12}$ and $\langle\Gamma_1,\Gamma_2\rangle$ have opposite sign. Returning to (5.85), we see that when the bound state is allowed, the pair of 1/2-BPS charges contribute to the Fourier coefficient at leading order with measure factor $\bar{c}(\Gamma_1)\bar{c}(\Gamma_2)|\langle\Gamma_1,\Gamma_2\rangle|$.

In contrast, when $[\hat{B}^\intercal\Omega_2^\star\hat{B}]_{12}$ and $\langle\Gamma_1,\Gamma_2\rangle$ have the same sign, the bound state is not allowed and the last term in (5.85) vanishes at the saddle point $\Omega_2 = \Omega_2^\star$ in (5.31). This term still contributes to the integral (5.72), but is exponentially suppressed. At large $R$, the integral is now dominated by the boundary of the chamber where the sign of $[\hat{B}^\intercal\Omega_2\hat{B}]_{12}$ flips, as shown in Appendix F.1. On this locus, the argument of the exponential $\text{Tr}\big[\frac{R^2}{S_2}\Omega_2^{-1}\begin{pmatrix} 1 & S_1 \\ S_1 & |S|^2 \end{pmatrix} + 2\Omega_2\begin{pmatrix} Q_R^2 & Q_R P_R \\ Q_R P_R & P_R^2 \end{pmatrix}\big]$ in (5.72) decomposes into two pieces associated to $\Gamma_1, \Gamma_2$,

$$\frac{R^2}{\sigma_2 S_2}\big[\hat{B}^\intercal\begin{pmatrix} 1 & S_1 \\ S_1 & |S|^2 \end{pmatrix}\hat{B}\big]_{11} + 2\sigma_2([\hat{B}^{-1}\Gamma_R]_1)^2 + \frac{R^2}{\rho_2 S_2}\big[\hat{B}^\intercal\begin{pmatrix} 1 & S_1 \\ S_1 & |S|^2 \end{pmatrix}\hat{B}\big]_{22} + 2\rho_2([\hat{B}^{-1}\Gamma_R]_2)^2\,. \tag{5.91}$$

---

[21]Supersymmetry implies that $Q_{iR}$ and $P_{iR}$ are linear combinations of $Q_R$ and $P_R$, but this is automatically the case for 1/2-BPS charges such that $Q_i\wedge P_i = 0$.

The integral is then exponentially suppressed by $e^{-2\pi R(\mathcal{M}(\Gamma_1)+\mathcal{M}(\Gamma_2))}$. The same holds for the contribution of the Dirac delta function which is computed explicitly in Appendix D.

We conclude that (5.72) receives contributions of each possible splitting $\Gamma = \Gamma_1 + \Gamma_2$, weighted by the product of the 1/2-BPS measures $\bar{c}(\Gamma_1)\bar{c}(\Gamma_2)$ and further exponentially suppressed by $e^{-2\pi R(\mathcal{M}(\Gamma_1)+\mathcal{M}(\Gamma_2))}$. It is important to distinguish these two-instanton contributions from one-instanton contributions due to bound states of 1/2-BPS states. Due to the triangular inequality $\mathcal{M}(\Gamma_1) + \mathcal{M}(\Gamma_2) \geq \mathcal{M}(\Gamma)$, these contributions are subdominant compared to the one-instanton contributions (5.77) away from the walls of marginal stability. On the wall, the two contributions become comparable and the complete Fourier coefficient is continuous.

This discussion generalizes with some efforts to CHL models with $N$ prime. In Appendix C we show that the measure function for $|\Omega_2| \geq \frac{1}{4}$ decomposes as

$$
\begin{aligned}
\bar{C}_{k-2}(Q,P;\Omega_2) = &\sum_{\substack{A\in M_2(\mathbb{Z})/GL(2,\mathbb{Z}) \\ A^{-1}\binom{Q}{P}\in\Lambda_m\oplus\Lambda_m}} |A| C_{k-2}^F\Big[A^{-1}\big(\begin{smallmatrix} -Q^2 & -Q\cdot P \\ -Q\cdot P & -P^2 \end{smallmatrix}\big)A^{-\intercal}\Big] \\
&+ \sum_{\substack{A\in M_{2,0}(N)/[\mathbb{Z}_2\ltimes\Gamma_0(N)] \\ A^{-1}\binom{Q}{P}\in\Lambda_m^*\oplus\Lambda_m}} |A| \widetilde{C}_{k-2}^F\Big[A^{-1}\big(\begin{smallmatrix} -Q^2 & -Q\cdot P \\ -Q\cdot P & -P^2 \end{smallmatrix}\big)A^{-\intercal}\Big] \\
&+ \sum_{\substack{A\in M_2(\mathbb{Z})/GL(2,\mathbb{Z}) \\ A^{-1}\binom{Q}{P/N}\in\Lambda_m^*\oplus\Lambda_m^*}} |A| C_{k-2}^F\Big[A^{-1}\big(\begin{smallmatrix} -NQ^2 & -Q\cdot P \\ -Q\cdot P & -P^2/N \end{smallmatrix}\big)A^{-\intercal}\Big] \\
&+ \sum_{\substack{\Gamma_i\in\Lambda_m^*\oplus\Lambda_m \\ Q_i\wedge P_i=0,\,\Gamma_1+\Gamma_2=\Gamma}} \bar{c}_k(\Gamma_1)\bar{c}_k(\Gamma_2)\bigg(-\frac{\delta([\hat{B}^\intercal\Omega_2\hat{B}]_{12})}{4\pi} + \frac{\langle\Gamma_1,\Gamma_2\rangle}{2}\big(\mathrm{sign}(\langle\Gamma_1,\Gamma_2\rangle)-\mathrm{sign}([\hat{B}^\intercal\Omega_2\hat{B}]_{12})\big)\bigg),
\end{aligned}
$$

(5.92)

with $\hat{B} \in SL(2,\mathbb{Q})/\mathrm{Stab}(\pi_i,\mathbb{Q})$ such that $\Gamma_i = \hat{B}\pi_i\hat{B}^{-1}\Gamma$. In this case one must distinguish the charges $\Gamma_1$ and $\Gamma_2$ that are twisted or untwisted to reproduce the exact measure (2.22). In Appendix C we analyze all the possible splittings depending on the orbit – electric or magnetic – of the charges $\Gamma_1$ and $\Gamma_2$ under $\Gamma_0(N)$. The sign $(-1)^{Q\cdot P} = (-1)^{\langle\Gamma_1,\Gamma_2\rangle}$ for all splittings, which ensures that the contribution of the sign function in (5.92) to the helicity supertrace $\Omega_6(Q,P,t)$ satisfies to the wall-crossing formula (2.12) with the correct sign.

It is interesting to understand this property from the differential equation imposed by supersymmetry Ward identities (2.26). We show explicitly in Appendix E.3 that the component of the differential equation with all indices along the decompactified torus is satisfied. In general, one finds that the leading contribution to the Fourier coefficient (5.72) with constant measure $\bar{C}_{k-2}(Q,P;\Omega_2) \sim \bar{C}_{k-2}(Q,P;\Omega_2^\star)$ as in (5.77), solves the homogeneous equation (3.17). The contributions due to the discontinuities of the summation measure $\bar{C}_{k-2}(Q,P;\Omega_2)$ give a particular inhomogeneous solution sourced by the quadratic term in $F_{abcd}$. For a given 1/4-BPS charge $\Gamma$, the Fourier coefficients of $F_{abcd}$ contribute a source term proportional to $\bar{c}_k(\Gamma_1)\bar{c}_k(\Gamma_2)$ for all possible splittings $\Gamma = \Gamma_1 + \Gamma_2$, which matches the structure of the measure measure in (5.92). In this way, the differential equation constrains the measure function to be consistent with wall crossing, such that the discontinuities must correspond to the sum over all possible splittings weighted by the 1/2-BPS measures of the constituent charges as exhibited in (5.92).

The explicit check of the differential equation in Appendix E.3 demonstrates that the unfolding procedure reproduces the correct Abelian Fourier coefficients, at least up to terms that are exponentially suppressed in $e^{-2\pi R^2}$. This is an important consistency check because the same unfolding procedure fails to reproduce the non-perturbative contributions to the constant terms associated to instanton anti-instantons, which are also required to be present in

order for the differential equation to hold . These effects are also necessary in order to re-solve the ambiguity of the sum over 1/4-BPS instantons [49], which is divergent due to the exponential growth of the measure $\bar{C}_{k-2}(Q,P;\Omega_2^\star) \sim (-1)^{Q\cdot P+1} e^{\pi|Q\wedge P|}$ [2,8].

# 6 Weak coupling expansion in dual string vacua

In section §4.3, we analyzed the weak coupling expansion of the exact $\nabla^2(\nabla\phi)^4$ in $D = 3$, in the limit where the heterotic string coupling is small. However, the CHL vacua of interest in this paper also admit dual descriptions in terms of freely acting orbifolds of type II string theory compactified on $K_3 \times T^3$ [73,74], or of type I strings on $T^7$ [75,76]. In this section, we discuss the weak coupling expansion of these exact results on the type II and type I sides. We also include a brief discussion of the $\nabla^2\mathcal{H}^4$ couplings in type IIB string theory compactified on $K3$, whose exact form, as conjectured in [47], involves the same type of genus-two modular integral, albeit with a lattice of signature $(21,5)$.

## 6.1 Weak coupling limit in CHL orbifolds of type II strings on $K3 \times T^3$

On the type II side, string vacua with 16 supercharges can be obtained by orbifolding the type II string on $K3 \times T^3$ by a symplectic automorphism of K3 combined with a translation on $T^3$ [73,74]. In order to keep manifest the four-dimensional origin of these models, we shall assume that the translation acts only on a $T^2$ inside $T^3$. In the weak coupling limit $g_6 \to 0$ (where $g_6$ is the string coupling in type IIA compactified on $K3$), the 'non-perturbative Narain lattice' (2.3) decomposes into [77],

$$\Lambda_{2k,8} \to \Lambda_{2k-4,4} \oplus \left[ I\!I_{1,1} \oplus I\!I_{1,1}[N] \right] \oplus \left[ I\!I_{1,1} \oplus I\!I_{1,1}[N] \right], \tag{6.1}$$

where the first summand is the sublattice of the homology lattice $\Lambda_{20,4} = H_{\text{even}}(K3)$ which is invariant under the symplectic automorphism, the second is the lattice of windings and momenta along $T^2$, and the third is the lattice of windings and momenta along $S_1$ together with the non-perturbative direction. The last two summands can be combined into a lattice $\Lambda_{4,4} = I\!I_{2,2} \oplus I\!I_{2,2}[N]$ which can be thought as the lattice of windings and momenta along a fiducial torus $T^4$. Assuming for simplicity that flat metric on the torus $T^3$ is diagonal and the Kalb-Ramond two-form vanishes, the radii of the four circles in this fiducial $T^4$ are related to the three radii $R_5, R_6, R_7$ of the physical $T^3$ by

$$(r_1, r_2, r_3, r_4) = \left( \frac{R_6}{g_6 \ell_{II}}, \frac{R_7}{g_6 \ell_{II}}, \frac{R_5}{g_6 \ell_{II}}, \frac{R_5 R_6 R_7}{g_6 \ell_{II}^3} \right). \tag{6.2}$$

In the limit $g_6 \to 0$, the four radii $r_i$ scale to infinity at the same rate, so the automorphism group $O(\Lambda_{4,4})$ is broken to a congruence subgroup of $SL(4,\mathbb{Z})$, which is identified with the T-duality group $O(\Lambda_{3,3})$ along the three-torus. In order to make T-duality invariance manifest, it is useful to define the type II string coupling in three-dimensions $g_3' = g_6 \sqrt{\ell_{II}^3/V_3}$ where $\ell_{II}$ is the type II string length and $V_3 = R_5 R_6 R_7$.

The analysis in §4.1 and §5.1 – and our previous analysis of the one-loop integral in [22] is readily generalized to the case where $n$ radii of a lattice $I\!I_{n-r,n-r} \oplus I\!I_{r,r}[N]$ become large,

leading in the maximal rank case $N = 1$ to

$$F^{(p,q)}_{\alpha\beta\gamma\delta} = V_n F^{(p-n,q-n)}_{\alpha\beta\gamma\delta} + \frac{3c(0)}{16\pi^2} V_n^{\frac{q-6}{n}} \Gamma(\tfrac{n+6-q}{2}) \sideset{}{'}\sum_{m^i \in \mathbb{Z}^n} (\pi\, m^i U_{ij} m^j)^{\frac{q-n-6}{2}} \delta_{(\alpha\beta}\delta_{\gamma\delta)} + \dots, \quad (6.3)$$

$$G^{(p,q)}_{\alpha\beta,\gamma\delta} = V_n^2 G^{(p-n,q-n)}_{\alpha\beta,\gamma\delta} - \frac{c(0)}{32\pi} V_n^{\frac{q+n-6}{n}} \Gamma(\tfrac{n+6-q}{2}) \sideset{}{'}\sum_{m^i \in \mathbb{Z}^n} (\pi\, m^i U_{ij} m^j)^{\frac{q-n-6}{2}} \delta_{\langle\alpha\beta,} G^{(p-n,q-n)}_{\gamma\delta\rangle}$$

$$- \frac{3}{4\pi}\left[ \frac{c(0)}{8\pi} V_n^{\frac{q-6}{n}} \Gamma(\tfrac{n+6-q}{2}) \sideset{}{'}\sum_{m^i \in \mathbb{Z}^n} (\pi m^i U_{ij} m^j)^{\frac{q-n-6}{2}} \right]^2 \delta_{\langle\alpha\beta,}\delta_{\gamma\delta\rangle} + \dots, \qquad (6.4)$$

or in the case of $N \neq 1$,

$$F^{(p,q)}_{\alpha\beta\gamma\delta} = V_n F^{(p-n,q-n)}_{\alpha\beta\gamma\delta} + \frac{3c_k(0)}{16\pi^2} V_n^{\frac{q-6}{n}} \Gamma(\tfrac{n+6-q}{2}) \delta_{(\alpha\beta}\delta_{\gamma\delta)} \left[ \sideset{}{'}\sum_{m^i \in \mathbb{Z}^n} (\pi\, m^i U_{ij} m^j)^{\frac{q-n-6}{2}} \right.$$

$$\left. + N^{r-1} \sideset{}{'}\sum_{\substack{m^1,\dots,m^{n-r}\in\mathbb{Z}^{n-r} \\ m^{n-r},\dots,m^n\in N\mathbb{Z}^r}} (\pi\, m^i U_{ij} m^j)^{\frac{q-n-6}{2}} \right] + \dots,$$

$$G^{(p,q)}_{\alpha\beta,\gamma\delta} = V_n^2 G^{(p-n,q-n)}_{\alpha\beta,\gamma\delta} - \frac{c_k(0)}{32\pi(N-1)} V_n^{\frac{q+n-6}{n}} \Gamma(\tfrac{n+6-q}{2}) \delta_{\langle\alpha\beta,}$$

$$\times \left[ \left( N^r \sideset{}{'}\sum_{\substack{m^1,\dots,m^{n-r}\in\mathbb{Z}^{n-r} \\ m^{n-r+1},\dots,m^n\in N\mathbb{Z}^r}} (\pi\, m^i U_{ij} m^j)^{\frac{q-n-6}{2}} - \sideset{}{'}\sum_{m^i\in\mathbb{Z}^n} (\pi\, m^i U_{ij} m^j)^{\frac{q-n-6}{2}} \right) G^{(p-n,q-n)}_{\gamma\delta\rangle} \right.$$

$$\left. + \left( N \sideset{}{'}\sum_{m^i\in\mathbb{Z}^n} (\pi\, m^i U_{ij} m^j)^{\frac{q-n-6}{2}} - N^{r-1}\sideset{}{'}\sum_{\substack{m^1,\dots,m^{n-r}\in\mathbb{Z}^{n-r} \\ m^{n-r+1},\dots,m^n\in N\mathbb{Z}^r}} (\pi\, m^i U_{ij} m^j)^{\frac{q-n-6}{2}} \right) \varsigma G^{(p-n,q-n)}_{\gamma\delta\rangle} \right] \qquad (6.5)$$

$$- \frac{3c_k(0)^2}{256\pi^3} V_n^{\frac{2q-12}{n}} \Gamma(\tfrac{n+6-q}{2})^2$$

$$\times \left[ \sideset{}{'}\sum_{m^i\in\mathbb{Z}^n} (\pi m^i U_{ij} m^j)^{\frac{q-n-6}{2}} + N^{r-1}\sideset{}{'}\sum_{\substack{m^1,\dots,m^{n-r}\in\mathbb{Z}^{n-r} \\ m^{n-r+1},\dots,m^n\in N\mathbb{Z}^r}} (\pi m^i U_{ij} m^j)^{\frac{q-n-6}{2}} \right]^2 \delta_{\langle\alpha\beta,}\delta_{\gamma\delta\rangle}$$

$$+ \frac{18 V_n^{\frac{2q-10}{n}}}{(N^2-1)\pi^{3/2}} \Gamma(\tfrac{n+5-q}{2})\Gamma(\tfrac{n+4-q}{2}) \delta_{\langle\alpha\beta,}\delta_{\gamma\delta\rangle}$$

$$\times \left( N \sideset{}{'}\sum_{\substack{A\in \\ M_{n,2}(\mathbb{Z})/GL(2,\mathbb{Z})}} - N^{r-1} \sideset{}{'}\sum_{\substack{A\in \\ M_{n,2,0}[N^r]/(\mathbb{Z}_2\ltimes\Gamma_0(N))}} + N^{2r-2}\sideset{}{'}\sum_{\substack{A\in \\ M_{n,2,00}[N^r]/GL(2,\mathbb{Z})}} \right) \det(\pi A^\intercal U A)^{\frac{q-n-5}{2}} + \dots,$$

where the dots denote exponentially suppressed terms and $U_{ij}$ is the metric on the $n$-torus, normalized to have unit determinant.[22] Here $M_{n,2}(\mathbb{Z})$ is the set of rank two $n$ by 2 matrices over the integers, $M_{n,2,0}[N^r]$ the subset for which the first column last $r$ entries vanish mod $N$, and $M_{n,2,00}[N^r]$ the subset for which the two columns last $r$ entries vanish mod $N$.

The sums over $m^i \in \mathbb{Z}^n\backslash\{0\}$ can be expressed in terms of the vector Eisenstein series for the congruence subgroup of $SL(n,\mathbb{Z})$ for which the lower left $r \times (n-r)$ entries vanish mod N in the fundamental matrix representation, which we denote by $SL_n[N^r]$,

$$\mathcal{E}^{\star SL_n[N^r]}_{s\Lambda_1}(U) = \frac{1}{2}\Gamma(s) \sideset{}{'}\sum_{\substack{m^1,\dots,m^{n-r+1}\in\mathbb{Z}^{n-r} \\ m^{n-r},\dots,m^n\in N\mathbb{Z}^r}} (\pi\, m^i U_{ij} m^j)^{-s}. \qquad (6.6)$$

---

[22]In the case of a square torus of volume $V_n = r_1\dots r_n$, $U_{ij} = r_i^2 \delta_{ij}/V_n^{\frac{2}{n}}$.

The sums over $A$ can be expressed in terms of rank two tensor Eisenstein series for the same congruence subgroup $SL_n[N^r]$

$$
\begin{aligned}
\mathcal{E}^{\star SL_n}_{s\Lambda_2}(U) &= \pi^{\frac{1}{2}}\Gamma(s)\Gamma(s-\tfrac{1}{2}) \sum_{\substack{A\in \\ M_{n,2}(\mathbb{Z})/GL(2,\mathbb{Z})}}^{\prime} \det(\pi A^{\mathsf{T}}UA)^{-s}\,, \\
\mathcal{E}^{\star SL_n[N^r]}_{s\Lambda_2,0}(U) &= \pi^{\frac{1}{2}}\Gamma(s)\Gamma(s-\tfrac{1}{2}) \sum_{\substack{A\in \\ M_{n,2,0}[N^r]/(\mathbb{Z}_2\ltimes\Gamma_0(N))}}^{\prime} \det(\pi A^{\mathsf{T}}UA)^{-s}\,, \\
\mathcal{E}^{\star SL_n[N^r]}_{s\Lambda_2,00}(U) &= \pi^{\frac{1}{2}}\Gamma(s)\Gamma(s-\tfrac{1}{2}) \sum_{\substack{A\in \\ M_{n,2,00}[N^r]/GL(2,\mathbb{Z})}}^{\prime} \det(\pi A^{\mathsf{T}}UA)^{-s}\,.
\end{aligned}
\tag{6.7}
$$

Note that for $N=1$, $\mathcal{E}^{\star SL_n}_{s\Lambda_k}(U)$ is the standard Langlands Eisenstein series satisfying the functional relation $\mathcal{E}^{\star SL_n}_{(\frac{n}{2}-s)\Lambda_k}(U) = \mathcal{E}^{\star SL_n}_{s\Lambda_k}(U^{-1})$.

For $(n,r)=(1,0)$ and $(n,r)=(2,1)$, (6.3) and (6.5) reduce to the results in §4 and 5 of [22] and the present paper, respectively. The case relevant in the present context is $(n,r)=(4,2)$. Setting $(p,q,n)=(2k,8,4)$, $V_4 = V_3^2/(g_6^4\ell_{II}^6) = 1/g_3^{\prime 4}$, and multiplying by a suitable power of $g_3'$ for translating to the string frame, we find that the perturbative terms in the $(\nabla\phi)^4$ and $\nabla^2(\nabla\phi)^4$ couplings in the maximal rank case are given by

$$
\begin{aligned}
g_3^{\prime 2} F^{(24,8)}_{\alpha\beta\gamma\delta} &= \frac{1}{g_3^{\prime 2}} F^{(20,4)}_{\alpha\beta\gamma\delta} + \frac{9}{\pi^2}\mathcal{E}^{\star SL_4}_{\Lambda_1}(U)\delta_{(\alpha\beta}\delta_{\gamma\delta)} + \dots\,, \\
g_3^{\prime 6} G^{(24,8)}_{\alpha\beta,\gamma\delta} &= \frac{1}{g_3^{\prime 2}} G^{(20,4)}_{\alpha\beta,\gamma\delta} - \frac{3}{2\pi}\mathcal{E}^{\star SL_4}_{\Lambda_1}(U)\delta_{\langle\alpha\beta,}G^{(20,4)}_{\gamma\delta\rangle} - \frac{27g_3^{\prime 2}}{\pi^3}[\mathcal{E}^{\star SL_4}_{\Lambda_1}(U)]^2\delta_{\langle\alpha\beta,}\delta_{\gamma\delta\rangle} + \dots\,.
\end{aligned}
\tag{6.8}
$$

Similarly, for $N>1$ we get

$$
\begin{aligned}
g_3^{\prime 2} F^{(2k,8)}_{\alpha\beta\gamma\delta} &= \frac{1}{g_3^{\prime 2}} F^{(2k-4,4)}_{\alpha\beta\gamma\delta} + \frac{9}{\pi^2(N+1)}\left[\mathcal{E}^{\star SL_4}_{\Lambda_1}(U) + N\mathcal{E}^{\star SL_4[N^2]}_{\Lambda_1}(U)\right]\delta_{(\alpha\beta}\delta_{\gamma\delta)} + \dots\,, \\
g_3^{\prime 6} G^{(2k,8)}_{\alpha\beta,\gamma\delta} &= \frac{1}{g_3^{\prime 2}} G^{(2k-4,4)}_{\alpha\beta,\gamma\delta} - \frac{3}{2\pi(N^2-1)}\left[N^2\mathcal{E}^{\star SL_4[N^2]}_{\Lambda_1}(U) - \mathcal{E}^{\star SL_4}_{\Lambda_1}(U)\right]\delta_{\langle\alpha\beta,}G^{(2k-4,4)}_{\gamma\delta\rangle} \\
&\quad - \frac{3N}{2\pi(N^2-1)}\left[\mathcal{E}^{\star SL_4}_{\Lambda_1}(U) - \mathcal{E}^{\star SL_4[N^2]}_{\Lambda_1}(U)\right]\delta_{\langle\alpha\beta,}{}^{\varsigma}G^{(2k-4,4)}_{\gamma\delta\rangle} \\
&\quad + \frac{18N}{(N^2-1)\pi^2}\delta_{\langle\alpha\beta,}\delta_{\gamma\delta\rangle}\left[\mathcal{E}^{\star SL_4}_{\frac{1}{2}\Lambda_2}(U) - \mathcal{E}^{\star SL_4[N^2]}_{\frac{1}{2}\Lambda_2,0}(U) + N\mathcal{E}^{\star SL_4[N^2]}_{\frac{1}{2}\Lambda_2,00}(U)\right] \\
&\quad - \frac{27g_3^{\prime 2}}{\pi^3(N+1)^2}\left[\mathcal{E}^{\star SL_4}_{\Lambda_1}(U) + N\mathcal{E}^{\star SL_4[N^2]}_{\Lambda_1}(U)\right]^2\delta_{\langle\alpha\beta,}\delta_{\gamma\delta\rangle} + \dots\,.
\end{aligned}
\tag{6.10}
$$

In either case, the rank 0, rank-1 and rank-2 orbits are now interpreted on the type II side as tree-level, one-loop and two-loop contributions, with an additional one-loop contribution in the rank-2 orbit for $N>1$. The tree-level contributions are consistent with the observation in [78] that the tree-level $F^4$ coupling of four twisted gauge bosons is governed by a genus-one modular integral, and the analogous statement in [79] that the tree-level $\nabla^2 F^4$ coupling of four twisted gauge bosons is governed by a genus-two modular integral. For $N=1$, the one-loop contributions are proportional to the vector Eisenstein series of $SL(4,\mathbb{Z})$, or equivalently the spinor Eisenstein series under the T-duality group $O(3,3)$ of the torus $T^3$, while the two-loop contribution is proportional to the square of the same. For $N>1$ they are similar

generalizations of Eisenstein series of $SL_4[N^2]$, and there is an additional contribution at 1-loop in rank two Eisenstein series of $SL_4[N^2]$, that are linear combinations of vector Eisenstein series of the group $O(3,3)$ of automorphisms of $II_{2,2} \oplus II_{1,1}[N]$.[23]

It would be interesting to confirm these predictions by independent one-loop and two-loop computations in type II string theory. Finally, the exponentially suppressed terms in (6.8) can be ascribed to D-brane, NS5-branes and KK (6,1)-brane instantons as explained in more detail in [78].

## 6.2 Weak coupling limit in type II string theory compactified on $K3 \times T^2$

Let us now consider the expansion of the exact $\nabla^2 F^4$ and $\mathcal{R}^2 F^2$ terms in $D = 4$ obtained in (5.70) at weak coupling on the type II side. Recall that the heterotic axiodilaton $S$ corresponds respectively to the 2-torus Kähler modulus $T_A$ in type IIA, and the 2-torus complex structure modulus $U_B$ in type IIB, while the type II axiodilaton $S_A = S_B$ corresponds to the Kähler modulus $T$ of the 2-torus on the heterotic side, *i.e.*

$$S = T_A = U_B \,, \qquad T = S_A = S_B \,, \qquad U = U_A = T_B \,. \tag{6.11}$$

In order to expand at small type II string coupling, *i.e.* at large $T_2$, we decompose the lattice $\Lambda_{2k-2,6}$ into $\Lambda_{2k-4,4} \oplus II_{1,1} \oplus II_{1,1}[N]$ as in section 5.2.

For simplicity we shall use the type IIB moduli in this section, and we won't write explicitly the label B. So $S$ is now the type IIB axiodilaton with $S_2 = \frac{1}{g_s^2}$. For simplicity we shall only consider the perturbative terms for the Maxwell fields in the RR sector, corresponding to indices $\alpha, \beta, \ldots$ along the sublattice $\Lambda_{2k-4,4}$. Using the results of [22], the perturbative part of the exact $F^4$ coupling is given by

$$\widehat{F}^{(2k-2,6)}_{\alpha\beta\gamma\delta \, II} = \frac{1}{g_s^2} F^{(2k-4,4)}_{\alpha\beta\gamma\delta \, II} + \frac{3}{2\pi} \delta_{(\alpha\beta}\delta_{\gamma\delta)} \left( \frac{\hat{\mathcal{E}}_1(NT) + \hat{\mathcal{E}}_1(T) + \hat{\mathcal{E}}_1(NU) + \hat{\mathcal{E}}_1(U) + \frac{12}{\pi}\log g_s}{N+1} \right)$$

$$= S_2 F^{(2k-4,4)}_{\alpha\beta\gamma\delta}(t) - \frac{3}{8\pi^2} \delta_{(\alpha\beta}\delta_{\gamma\delta)} \log(S_2^k T_2^k U_2^k |\Delta_k(T)\Delta_k(U)|^2) \,, \tag{6.12}$$

where the first term matches the tree-level coupling computed in [78], while the second term is related by supersymmetry to the $\mathcal{R}^2$ coupling computed in [80, 81].

The exact $\nabla^2 F^4$ coupling is obtained from (5.70) after dropping the logarithmic terms in $R$,

$$\widehat{G}^{(2k-2,6)}_{ab,cd \, NP}(U,\varphi) = \widehat{G}^{(2k-2,6)}_{ab,cd}(\varphi) - \frac{3}{4\pi} \delta_{\langle ab,}\delta_{cd\rangle} \left( \frac{\hat{\mathcal{E}}_1(NU) + \hat{\mathcal{E}}_1(U)}{N+1} \right)^2 \tag{6.13}$$

$$- \frac{1}{4} \delta_{\langle ab,} \left( \frac{N\hat{\mathcal{E}}_1(NU) - \hat{\mathcal{E}}_1(U)}{N^2-1} \widehat{G}^{(2k-2,6)}_{cd\rangle}(\varphi) + \frac{N\hat{\mathcal{E}}_1(U) - \hat{\mathcal{E}}_1(NU)}{N^2-1} \varsigma\widehat{G}^{(2k-2,6)}_{cd\rangle}(\varphi) \right) \,,$$

where $U$ parametrizes $SL(2)/SO(2)$ and $\varphi$ the Grassmannian on $\Lambda_{2k-2,6}$. The power-behaved term of $\widehat{G}^{(2k-2,6)}_{ab,cd}(\varphi)$ in this limit is given in equations (5.36), (4.59) and (5.60) for $q = 6$, $v = N$, $R = \sqrt{S_2} = \frac{1}{g_s}$, and $\varphi = t$ the K3 moduli of the Grassmanian $G_{(2k-4,4)}$. After expanding around

---

[23]The condition that $SL(4,\mathbb{Z})$ preserves the lattice $II_{2,2} \oplus II_{1,1}[N]$, so $Q_{34} = 0[N]$, implies that the matrices are either of type $\left(\begin{smallmatrix} \cdot & \cdot & \cdot & \cdot \\ \cdot & \cdot & \cdot & \cdot \\ 0 & 0 & \cdot & \cdot \\ 0 & 0 & \cdot & \cdot \end{smallmatrix}\right)$ mod $N$ or of type $\left(\begin{smallmatrix} \cdot & \cdot & \cdot & \cdot \\ \cdot & \cdot & \cdot & \cdot \\ \cdot & \cdot & \cdot & \cdot \\ 0 & 0 & 0 & 0 \end{smallmatrix}\right)$ mod $N$, but the condition that the it preserves the dual lattice, *i.e.* $Q_{ij} \in \mathbb{Z}$ for $ij \neq 12$ with $NQ_{12} \in \mathbb{Z}$ forbids the second.

$q = 6 + 2\epsilon$ and subtracting polar terms,[24] we find

$$
\widehat{G}^{(2k-2,6)}_{\alpha\beta,\gamma\delta}(\varphi) \sim \frac{1}{g_s^4} \widehat{G}^{(2k-4,4)}_{\alpha\beta,\gamma\delta}(t) - \frac{3}{4\pi} \delta_{\langle\alpha\beta} \delta_{\gamma\delta\rangle} \Big( \frac{\hat{\mathcal{E}}_1(NT) + \hat{\mathcal{E}}_1(T) + \frac{12}{\pi}\log g_s}{N+1} \Big)^2
$$
$$
- \frac{1}{4g_s^2} \delta_{\langle\alpha\beta,} \Big( \frac{\frac{N\hat{\mathcal{E}}_1(NT) - \hat{\mathcal{E}}_1(T)}{N-1} + \frac{6}{\pi}\log g_s}{N+1} \widehat{G}^{(2k-4,4)}_{\gamma\delta\rangle}(t) + \frac{\frac{N\hat{\mathcal{E}}_1(T) - \hat{\mathcal{E}}_1(NT)}{N-1} + \frac{6}{\pi}\log g_s}{N+1} \,{}^{\varsigma}\widehat{G}^{(2k-4,4)}_{\gamma\delta\rangle}(t) \Big).
$$
(6.14)

To compute the power-like term of $\widehat{G}^{(2k-2,6)}_{ab}(\varphi)$ one proceeds as in [22], and finds after expanding around $q = 6 + 2\epsilon$ and subtracting polar terms

$$
\widehat{G}^{(2k-2,6)}_{\alpha\beta}(\varphi) \sim \frac{1}{g_s^2} \Big( \widehat{G}^{(2k-4,4)}_{\alpha\beta}(t) + \frac{2N}{N+1} \delta_{\alpha\beta} \big( \hat{\mathcal{E}}_1(T) - \hat{\mathcal{E}}_1(NT) \big) \Big)
$$
$$
+ \frac{12}{N+1} \frac{1}{2\pi} \delta_{\alpha\beta} \Big( \frac{12}{\pi}\log(g_s) + \hat{\mathcal{E}}_1(T) + \hat{\mathcal{E}}_1(NT) \Big). \quad (6.15)
$$

The function $^{\varsigma}\widehat{G}^{(2k-2,6)}_{ab}(\varphi)$ is obtained by acting with the involution $\varsigma$ on the K3 moduli $t$ and on the Kähler moduli $T$ by Fricke duality $T \to -\frac{1}{NT}$, so that

$$
^{\varsigma}\widehat{G}^{(2k-2,6)}_{\alpha\beta}(\varphi) \sim \frac{1}{g_s^2} \Big( {}^{\varsigma}\widehat{G}^{(2k-4,4)}_{\alpha\beta}(t) + \frac{2N}{N+1} \delta_{\alpha\beta} \big( \hat{\mathcal{E}}_1(NT) - \hat{\mathcal{E}}_1(T) \big) \Big)
$$
$$
+ \frac{12}{N+1} \frac{1}{2\pi} \delta_{\alpha\beta} \Big( \frac{12}{\pi}\log(g_s) + \hat{\mathcal{E}}_1(T) + \hat{\mathcal{E}}_1(NT) \Big). \quad (6.16)
$$

Collecting all terms, we obtain the complete perturbative $\nabla^2 F^4$ coupling in $D = 4$,

$$
\widehat{G}^{(2k-2,6)}_{\alpha\beta,\gamma\delta\,\mathrm{II}} = \frac{1}{g_s^4} \widehat{G}^{(2k-4,4)}_{\alpha\beta,\gamma\delta}(t)
$$
$$
- \frac{1}{4(N+1)g_s^2} \delta_{\langle\alpha\beta,} \Big( \Big( \frac{N\hat{\mathcal{E}}_1(NT) - \hat{\mathcal{E}}_1(T) + N\hat{\mathcal{E}}_1(NU) - \hat{\mathcal{E}}_1(U)}{N-1} + \frac{6}{\pi}\log g_s \Big) \widehat{G}^{(2k-4,4)}_{\gamma\delta\rangle}(t)
$$
$$
+ \Big( \frac{N\hat{\mathcal{E}}_1(T) - \hat{\mathcal{E}}_1(NT) + N\hat{\mathcal{E}}_1(U) - \hat{\mathcal{E}}_1(NU)}{N-1} + \frac{6}{\pi}\log g_s \Big) {}^{\varsigma}\widehat{G}^{(2k-4,4)}_{\gamma\delta\rangle}(t)
$$
$$
- 2N\delta_{\gamma\delta\rangle} \frac{(\hat{\mathcal{E}}_1(T) - \hat{\mathcal{E}}_1(NT))(\hat{\mathcal{E}}_1(U) - \hat{\mathcal{E}}_1(NU))}{N-1} \Big)
$$
$$
- \frac{3}{4\pi} \delta_{\langle\alpha\beta,} \delta_{\gamma\delta\rangle} \Big( \frac{\hat{\mathcal{E}}_1(NT) + \hat{\mathcal{E}}_1(T) + \hat{\mathcal{E}}_1(NU) + \hat{\mathcal{E}}_1(U) + \frac{12}{\pi}\log g_s}{N+1} \Big)^2. \quad (6.17)
$$

The terms involving $\log g_s$ originate as usual from the mixing between the local and non-local terms in the effective action [82]. The result (6.17) is manifestly invariant under the exchange of $U$ and $T$, hence identical in type IIA and type IIB. It is also invariant under the combined Fricke duality $T \to -\frac{1}{NT}$, $U \to -\frac{1}{NU}$, $t \to \varsigma t$ [27], which is built in our conjecture for the non-perturbative amplitude. In the maximal rank case, (6.17) must be replaced by [25]

$$
G^{(22,6)}_{\alpha\beta,\gamma\delta\,\mathrm{II}} = \frac{1}{g_s^4} \widehat{G}^{(20,4)}_{\alpha\beta,\gamma\delta}(t) + \frac{3}{4\pi g_s^2} \delta_{\langle\alpha\beta,} \Big( \log(T_2|\eta(T)|^4) + \log(U_2|\eta(U)|^4) - 2\log g_s \Big) G^{(20,4)}_{\gamma\delta\rangle}(t)
$$
$$
- \frac{27}{4\pi^3} \delta_{\langle\alpha\beta} \delta_{\gamma\delta\rangle} \Big( \log(T_2|\eta(T)|^4) + \log(U_2|\eta(U)|^4) - 2\log g_s \Big)^2. \quad (6.18)
$$

---

[24]Note that the lattice is fixed to $\Lambda_{2k-2,6}$, and the expansion in $q = 6 + 2\epsilon$ only applies to the numerical value of the various exponents, just like if one introduced a regularizing factor of $|\Omega_2|^\epsilon$ in the genus 2 integral.

[25]Note that $G^{(20,4)}_{\alpha\beta}$ is finite for the maximal rank case, whereas $\widehat{G}^{(2k-4,4)}_{\alpha\beta}$ requires in general a regularization due to the 1-loop supergravity divergence in six dimensions.

It would be interesting to check these predictions by explicit perturbative computations in type II string theory. Noting that

$$\frac{\hat{\mathcal{E}}_1(NT) + \hat{\mathcal{E}}_1(T)}{N+1} = -\frac{1}{4\pi} \log(T_2^{\,k}|\Delta_k(T)|) , \qquad \hat{\mathcal{E}}_1(T) = -\frac{1}{4\pi} \log(T_2^{\,12}|\Delta(T)|) , \qquad (6.19)$$

the 2-loop contribution on the last line of (6.17) takes the suggestive form

$$-\frac{3}{(4\pi)^3} \delta_{\langle\alpha\beta}, \delta_{\gamma\delta\rangle} \big( \log(S_2^{\,k} T_2^{\,k} U_2^{\,k} |\Delta_k(T)\Delta_k(U)|^2) \big)^2 . \qquad (6.20)$$

The $(\log g_s)^2$ term is consistent with the 2-loop logarithmic divergence of the four-photon amplitude [83] (recall that the $\log g_s$ can be traced back to the logarithm of the Mandelstam variables in the full amplitude, and therefore to the logarithm supergravity divergences [22, 82]). The term linear in $\log g_s$ in (6.20), corresponding to the $t_8 F^4$ form factor divergence, can be rewritten as

$$-\frac{3k}{4\pi} \log g_s \delta_{\langle\alpha\beta}, \Big( \frac{1}{12 g_s^{\,2}} \big( \widehat{G}_{\gamma\delta\rangle}^{(2k-4,4)}(t) + {}^\varsigma \widehat{G}_{\gamma\delta\rangle}^{(2k-4,4)}(t) \big) - \delta_{\gamma\delta\rangle} \frac{1}{8\pi^2} \log(T_2^{\,k} U_2^{\,k} |\Delta_k(T)\Delta_k(U)|^2) \Big)$$

$$= -\frac{3}{4\pi} \log g_s \, \delta_{\langle\alpha\beta}, \Big( \frac{1}{g_s^{\,2}} F_{\gamma\delta\rangle\eta}^{(2k-4,4)\eta}(t) - \delta_{\gamma\delta\rangle} \frac{2k}{(4\pi)^2} \log(T_2^{\,k} U_2^{\,k} |\Delta_k(T)\Delta_k(U)|^2) \Big)$$

$$= -\frac{3}{4\pi} \log g_s \, \delta_{\langle\alpha\beta}, \widehat{F}_{\gamma\delta\rangle c \text{ II}}^{(2k-2,6)c} , \qquad (6.21)$$

where one uses integration by part on the definition of $F^{(2k-2,6)}$ with $-\frac{1}{i\pi} \frac{\partial}{\partial \tau} \frac{1}{\Delta_k(\tau)} = \frac{k}{12}(E_2(\tau) + N E_2(N\tau))/\Delta_k(\rho)$, and $\delta_{(ab}\delta_{cd)}\delta^{cd} = \frac{2k}{3}\delta_{ab}$. Ignoring these logarithmic contributions, the two-loop coupling (6.20) does not depend on the K3 moduli, as required by supersymmetry, and might be computable in topological string theory.

The amplitudes with two photons in the Ramond sector and two gravitons can be obtained in the same way. It is non vanishing only when the two photons have the same polarization and the two gravitons have the opposite polarization. In type IIB, the complex amplitude is obtained through the Kähler derivative of the same function (6.17) with respect to $U$, e.g. in the maximal rank case

$$R_{\alpha\beta \text{ II}}^{(22,6)} = -\frac{9}{2\pi^3} \delta_{\alpha\beta} \hat{E}_2(U) \Big( \log(T_2|\eta(T)|^4) + \log(U_2|\eta(U)|^4) - 2 \log g_s \Big) + \frac{1}{4\pi g_s^{\,2}} \hat{E}_2(U) G_{\alpha\beta}^{(20,4)}(t) , \qquad (6.22)$$

or with respect to $T$ in type IIA. The $\log g_s$ term can be interpreted as the divergence of the form factor of the operator $\mathcal{R} F_R^2$ (where $F_R^{\dot{\alpha}}$ are the graviphoton field strengths) belonging to the $\mathcal{R}^2$-type supersymmetric invariant.

## 6.3 Type I string theory

The heterotic string with gauge group $Spin(16)/\mathbb{Z}_2$ is dual to the type I superstring [84]. In ten dimensions, the duality inverts the string coupling $e^\phi \to e^{-\phi}$ and identifies the Einstein frame metrics. After compactifying on a torus $T^q$, the effective string coupling $g_s$ in $10-q$ dimensions and volume $V_s$ in string units are given by

$$g_s = e^{(1-\frac{q}{8})\phi} V^{-\frac{1}{2}} , \qquad V_s = e^{\frac{q}{4}\phi} V , \qquad (6.23)$$

where $V$ is the volume of the torus $T^q$ measured in ten-dimensional Planck units. It follows that the heterotic/type I duality identifies

$$g_s = g_s'^{-1+\frac{q}{4}} V_s'^{-1+\frac{q}{8}} , \qquad V_s = g_s'^{-\frac{q}{2}} V_s'^{1-\frac{q}{4}} , \qquad (6.24)$$

where the unprimed variables refer to the heterotic string while the primed variables refer to the type I string, the unit volume metric $U_{ij}$ being the same on both sides. In particular, the weak coupling regime $g'_s \to 0$ on the type I side corresponds to strong coupling on the heterotic side when $D = 10 - q > 6$, or to weak coupling when $D < 6$. In either case, the volume $V'_s$ in heterotic string units scales to infinity. Furthermore, in dimension $D > 4$ the coefficients of the $F^4$ and $\nabla^2 F^4$ couplings are purely perturbative on the heterotic side, so their type I dual expansion is obtained by taking the large volume limit. We shall now show that the resulting weak coupling expansion on the type I side has only powers of the form $g'^{2h+b-2}_s$, compatible with type I genus expansion where $b$ is the number of boundaries or crosscaps. For simplicity we focus on the maximal rank model and consider only gauge bosons with indices along the $D_{16}$ lattice, but these considerations easily extend to type I models with reduced rank [75,76] and to gauge bosons with indices along the torus.

Using (6.3) and similar computations using the same method, we find that for $D > 4$, the $F^4$ effective interaction at weak type I coupling $g'_s \to 0$ is given by

$$
g'^{2\frac{q-2}{8-q}}_s F^I_{\alpha\beta\gamma\delta} = \frac{V'^{\frac{1}{2}}_s}{g'_s} F^{(16,0)}_{\alpha\beta\gamma\delta} + \frac{3}{2\pi} g'_s V'^{\frac{3}{2}}_s \delta_{(\alpha\beta}\delta_{\gamma\delta)} + \frac{9}{\pi^2} g'^2_s V'^{2-\frac{6}{q}}_s \sum_{m^i \in \mathbb{Z}^n}{}' (\pi\, m^i U_{ij} m^j)^{-3} \delta_{(\alpha\beta}\delta_{\gamma\delta)}
$$

$$
+ \frac{V'^{1-\frac{2}{q}}_s}{\pi} \sum_{\substack{Q \in D_{16} \\ Q^2 = 2}} \sum_{m \in \mathbb{Z}^q}{}' e^{2\pi i m^i Q \cdot a_i} \left( \frac{Q_\alpha Q_\beta Q_\gamma Q_\delta}{m^i U_{ij} m^j} - \frac{3V'^{\frac{1}{2}-\frac{2}{q}}_s}{2\pi^2} g'_s \frac{\delta_{(\alpha\beta}Q_\gamma Q_{\delta)}}{(m^i U_{ij} m^j)^2} + \frac{3V'^{1-\frac{4}{q}}_s}{8\pi^4} g'^2_s \frac{\delta_{(\alpha\beta}\delta_{\gamma\delta)}}{(m^i U_{ij} m^j)^3} \right)
$$

$$
+ \dots, \quad (6.25)
$$

where the dots stand for non-perturbative corrections associated to D1 branes wrapping two-cycles inside $T^q$. The first term is the expected disk amplitude of 4 open string gauge bosons in type I, while the remaining terms of order $g'^0_s, g'^1_s, g'^2_s$ are contributions from open Riemann surfaces $\chi = 0, -1, -2$ [85] (recall that $\chi = 2 - 2h - b$ for a Riemann surface with $h$ handles and $b$ boundaries). Similarly, the $\nabla^2 F^4$ coupling reads

$$
g'^{\frac{2q}{8-q}}_s G^I_{\alpha\beta,\gamma\delta} = \frac{1}{g'^2_s} G^{(16,0)}_{\alpha\beta,\gamma\delta} - \frac{V'_s}{4} \delta_{\langle\alpha\beta}, G^{(16,0)}_{\gamma\delta\rangle} - \frac{3}{2\pi} g_s V'^{\frac{3}{2}-\frac{6}{q}}_s \sum_{m^i \in \mathbb{Z}^q}{}' (\pi\, m^i U_{ij} m^j)^{-3} \delta_{\langle\alpha\beta}, G^{(16,0)}_{\gamma\delta\rangle}
$$

$$
+ \frac{3}{4\pi} \left( -g'^2_s V'^2_s + \frac{2}{\pi^2} g'^3_s V'^{\frac{5}{2}-\frac{6}{q}}_s - \frac{1}{V'_s} \Big[ \frac{6}{\pi} g'^2_s V'^{2-\frac{6}{q}}_s \sum_{m^i \in \mathbb{Z}^q}{}' (\pi m^i U_{ij} m^j)^{-3} \Big]^2 \right) \delta_{\langle\alpha\beta}, \delta_{\gamma\delta\rangle}
$$

$$
- g'_s \frac{V'^{\frac{3}{2}-\frac{2}{q}}_s}{4\pi} \delta_{\langle\alpha\beta}, \sum_{\substack{Q \in D_{16} \\ Q^2 = 2 \\ m \in \mathbb{Z}^q \setminus \{0\}}} e^{2\pi i m^i Q \cdot a_i} \left( \frac{Q_\gamma Q_{\delta\rangle}}{m^i U_{ij} m^j} - \frac{g'_s V'^{\frac{1}{2}-\frac{2}{q}}_s}{4\pi^2} \frac{12 Q_\gamma Q_{\delta\rangle} - \delta_{\gamma\delta\rangle}}{(m^i U_{ij} m^j)^2} + \frac{g'^2_s}{8\pi^4} \frac{3V'^{1-\frac{4}{q}}_s \delta_{\gamma\delta\rangle}}{(m^i U_{ij} m^j)^3} \right)
$$

$$
+ 3 \sum_{\substack{Q \in D_{16} \\ Q^2 = 2}}{}' \bar{G}^{(16,0)}_{\langle\alpha\beta}, (Q) \sum_{m \in \mathbb{Z}^q}{}' e^{2\pi i m^i Q \cdot a_i} \left( Q_\gamma Q_{\delta\rangle} \frac{V'^{1-\frac{4}{q}}_s}{(\pi\, m^i U_{ij} m^j)^2} - \frac{g'_s}{2\pi} \delta_{\gamma\delta\rangle} \frac{V'^{\frac{3}{2}-\frac{6}{q}}_s}{(\pi\, m^i U_{ij} m^j)^3} \right)
$$

$$
+ g'_s \sum_{\substack{Q_i \in D_{16} \oplus D_{16} \\ Q_i^2 \le 2}}{}' \int_{\mathcal{P}_2} \frac{d^3\Omega_2}{|\Omega_2|^3} C(Q, \tfrac{1}{g'_s}\Omega_2) \sum_{A \in M_{q,2}(\mathbb{Z})/GL(2,\mathbb{Z})}{}' P_{\alpha\beta,\gamma\delta}(Q, \tfrac{1}{g'_s}\Omega_2) e^{2\pi i a \cdot A \cdot Q - \pi V'^{\frac{2}{q}-\frac{1}{2}}_s \mathrm{Tr}[A\Omega_2^{-1}A^\intercal U]}
$$

$$
+ \dots, \quad (6.26)
$$

where the dots stand for non-perturbative corrections associated to D1 branes wrapping two-cycles inside $T^q$. In the last term, the integral of the constant part $C^F(Q)$ of the Fourier coefficient of $1/\Phi_{10}$ produces a matrix-variate Gamma function and contributes to order $g'_s, g'^2_s, g'^3_s$.

The jumps in $C(Q, \frac{1}{g_s'} \Omega_2)$ dues to poles at large $|\Omega_2|$ give terms of order $g_s'^\ell$ for $\ell = 0, 1, 2, 3, 4$, which are sourced by the square of the 'Wilson lines corrections' in (6.25) in the differential equation (2.26). The jumps due to deep poles where $|\Omega_2| \leq \frac{1}{4}$ lead to further corrections of order $e^{-2\pi/g_s'}$, which can be ascribed to D1-anti-D1 instantons.

The first term $\frac{1}{g_s'^2} G^{(16,0)}_{\alpha\beta,\gamma\delta}$ in (6.26) is however apparently inconsistent with type I perturbation theory, since the four-photon amplitude only involves open string vertex operators which cannot couple at genus zero. Fortunately, we can show that this term vanishes for the heterotic $Spin(16)/\mathbb{Z}_2$ string. Indeed, using the same integration by parts argument as in section 3.3 (the boundaries at the cusp do not contribute at $q = 0$) one finds

$$20G^{(16,0)}_{\alpha\beta,\gamma\delta} + \delta_{\langle\alpha\beta,} G^{(16,0)\,\epsilon}_{\gamma\delta\rangle,\epsilon} = \pi F^{\epsilon\zeta\,(16,0)}_{\langle\alpha\beta,} F^{(16,0)}_{\gamma\delta\rangle,\epsilon\zeta} = 0 \,, \tag{6.27}$$

which vanishes because [22, (5.42)]

$$F^{(16,0)}_{\alpha\beta\gamma\delta} = 16\pi\delta_{\alpha\beta\gamma\delta} \,, \tag{6.28}$$

where $\delta_{\alpha\beta\gamma\delta}$ is equal to one if all for indices are equal and zero otherwise. It follows that

$$G^{(16,0)}_{\alpha\beta,\gamma\delta} = \text{R.N.} \int_{Sp(4,\mathbb{Z})\backslash\mathcal{H}_2} \frac{\mathrm{d}^3\Omega_1 \mathrm{d}^3\Omega_2}{|\Omega_2|^3} \frac{\Gamma^{(2)}_{D_{16}}[P_{\alpha\beta,\gamma\delta}]}{\Phi_{10}} = 0 \,, \tag{6.29}$$

so (6.26) is indeed consistent with type I perturbation theory. In particular, the genus-two double trace $\nabla^2(\mathrm{Tr}F^2)^2$ coupling computed in [44] for the ten-dimensional $Spin(16)/\mathbb{Z}_2$ heterotic string vanishes. It is worth stressing that the same genus-two coupling in the $E_8 \times E_8$ string does *not* vanish. [26]

Let us now discuss the form of the non-perturbative corrections in some more details. For any $D \geq 3$, the contributions of the non-Abelian rank-2 orbit are non-perturbative on the type I side, with an action given for vanishing gauge charge by

$$S_{D1} = 2\pi \frac{V_s'^{\frac{2}{q}}}{g_s' V_s'^{\frac{1}{2}}} \sqrt{\tfrac{1}{2} U_{ik} U_{jl} N^{ij} N^{kl}} + 2\pi \mathrm{i} B_{ij} N^{ij} \,, \tag{6.33}$$

where $g_s' V_s'^{\frac{1}{2}} = e^{\phi'}$ is the ten-dimensional type I string coupling. This can be ascribed to Euclidean D1 branes wrapping $T^q$ with charge $N^{ij} \in \mathbb{Z}^q \wedge \mathbb{Z}^q$. For $D = 4$, the NS5-brane instantons on the heterotic side translate into D5-brane instantons on the type I side, with action $S_2 = \frac{V_s'^{\frac{1}{2}}}{g_s'}$. For $D = 3$, the non-perturbative heterotic contributions with vanishing NUT charge translate into type I D5-brane instantons with wrapping number $N_i$ and gauge charge $Q \in D_{16}$, with action

$$\mathrm{Re}[S_{D5}] = 2\pi \frac{V_s'^{\frac{6}{7}}}{g_s' V_s'^{\frac{1}{2}}} \sqrt{(U^{-1})^{ij}(N_i + a_i \cdot Q)(N_j + a_j \cdot Q)} \,, \tag{6.34}$$

---

[26]For the $E_8 \times E_8$ heterotic string, we have instead

$$20G^{(16,0)}_{\alpha\beta,\gamma\delta} + \delta_{\langle\alpha\beta,} G^{(16,0)\epsilon}_{\gamma\delta\rangle,\epsilon} = \pi F^{\epsilon\zeta\,(16,0)}_{\langle\alpha\beta,} F^{(16,0)}_{\gamma\delta\rangle,\epsilon\zeta} = \frac{64\pi^3}{3}(4P^1_{\langle\alpha\beta,} P^1_{\gamma\delta\rangle} + 4P^2_{\langle\alpha\beta,} P^2_{\gamma\delta\rangle} - 7P^1_{\langle\alpha\beta,} P^2_{\gamma\delta\rangle}) \,, \tag{6.30}$$

with

$$F^{(16,0)}_{\alpha\beta\gamma\delta} = 8\pi(P^1_{(\alpha\beta} P^1_{\gamma\delta)} + P^2_{(\alpha\beta} P^2_{\gamma\delta)} - P^1_{(\alpha\beta} P^2_{\gamma\delta)}) \,, \tag{6.31}$$

and $P^i_{\alpha\beta}$ the two projectors to the eight-dimensional subspaces. One computes that $G^{(16,0)\gamma}_{\alpha\beta,\gamma} = 0$, such that

$$G^{(16,0)}_{\alpha\beta,\gamma\delta} = \int_{Sp(4,\mathbb{Z})\backslash\mathcal{H}_2} \frac{\mathrm{d}^3\Omega_1 \mathrm{d}^3\Omega_2}{|\Omega_2|^3} \frac{\Gamma^{(2)}_{E_8\oplus E_8}[P_{\alpha\beta,\gamma\delta}]}{\Phi_{10}} = \frac{16\pi^3}{15}(4P^1_{\langle\alpha\beta,} P^1_{\gamma\delta\rangle} + 4P^2_{\langle\alpha\beta,} P^2_{\gamma\delta\rangle} - 7P^1_{\langle\alpha\beta,} P^2_{\gamma\delta\rangle}) \,. \tag{6.32}$$

This reproduces the relative coefficient in [86, (7.4)].

Finally, non-perturbative heterotic instantons with non-vanishing NUT charge translate into type I Taub-NUT instantons, with action

$$\text{Re}[S_{TN}] = 2\pi \frac{V_s'^{\frac{8}{7}}}{g_s'^2 V_s'} \sqrt{U_{ij}(k^i + g_s' V_s'^{\frac{3}{14}}(U^{-1})^{ik}\tilde{N}_k)(k^j + g_s' V_s'^{\frac{3}{14}}(U^{-1})^{jl}\tilde{N}_l)} \,, \tag{6.35}$$

with

$$\tilde{N}_i = N_i + a_i \cdot Q + (\tfrac{1}{2}a_i \cdot a_j + B_{ij})k^j \,. \tag{6.36}$$

Thus, all non-perturbative effects on the heterotic side map to expected instanton effects in type I.

## 6.4   Exact $\nabla^2 \mathcal{H}^4$ couplings in type IIB on K3

Finally, let us briefly discuss the couplings of four self-dual three-form field strengths $\mathcal{H}^a_{\mu\nu\rho}$ in type IIB string theory compactified on K3. In [47, 78], it was conjectured that the exact $\mathcal{H}^4$ coupling is given by a genus-one modular integral of the form (1.4) for the non-perturbative Narain lattice $\Lambda_{21,5}$ of signature $(p, q) = (21, 5)$. This was later generalized to the case of the $\nabla^2 \mathcal{H}^4$ couplings, which were conjectured to be given exactly by a genus-two modular integral of the form (1.5) for the same lattice [47]. These conjectures follow from our exact non-perturbative results for the maximal rank model[27] in $D = 3$ by decompactification. Here, we briefly discuss the weak coupling expansion of these results on the type IIB side, using the results of section 4.1.

At weak coupling, the even self-dual lattice $\Lambda_{21,5}$ decomposes into $\Lambda_{20,4} \oplus II_{1,1}$, where the 'radius' associated to the second factor is related to the type IIB string coupling by $g_s = 1/R$. The low energy action in the string frame was recalled in [22, 4.40], after changing the metric for $\gamma = g_s \gamma_E$ and renormalising the Ramond-Ramond field as $\mathcal{H}^a = g_s H^a$. The coefficient of the $\nabla^2 \mathcal{H}^4$ coupling in this frame is then given by $G^{(21,5)}_{\alpha\beta,\gamma,\delta}$, without any further power of $g_s$. The results of section 4.1 then provide its weak coupling expansion,

$$\begin{aligned}
G^{(21,5)}_{\alpha\beta,\gamma\delta} = {} & \frac{1}{g_s^2} G^{(20,4)}_{\alpha\beta,\gamma\delta} - \frac{1}{4}\delta_{\langle\alpha\beta,} G^{(20,4)}_{\gamma\delta\rangle} - \frac{3g_s^2}{4\pi}\delta_{\langle\alpha\beta,}\delta_{\gamma\delta\rangle} \\
& + \frac{3}{g_s^4}\sideset{}{'}\sum_{Q\in\Lambda^*_{21,5}} e^{2\pi iQ\cdot a}\, \bar{G}^{(20,4)}_{\langle\alpha\beta,}(Q,\varphi)\Big(Q_{L\gamma}Q_{L\delta\rangle}\frac{K_0(\frac{2\pi}{g_s^2}\sqrt{2Q_R^2})}{\sqrt{2Q_R^2}} - \frac{g_s^2}{4\pi}\delta_{\gamma\delta\rangle}K_1(\tfrac{2\pi}{g_s^2}\sqrt{2Q_R^2})\Big) \\
& + \sideset{}{'}\sum_{Q\in\Lambda^*_{21,5}} e^{-\frac{4\pi}{g_s^2}\sqrt{2Q_R^2}} K_{\alpha\beta,\gamma\delta}(g_s, Q_L, Q_R).
\end{aligned} \tag{6.37}$$

The first term proportional to $G^{(20,4)}_{\alpha\beta,\gamma\delta}$ is recognized as a tree-level contribution in type IIB on $K3$ [79]. The second and third terms correspond to one-loop and two-loop corrections, and to our knowledge have not been computed independently yet. The second line of (6.37) corresponds to exponentially suppressed terms that originate from D3, D1, D(-1) branes wrapped on K3 [78], or, formally, to Fourier coefficients of the coupling coefficient. The function $\bar{G}^{(21,5)}_{\alpha\beta}$ is the sum of a finite and a polar contribution and reads

$$\bar{G}^{(20,4)}_{\alpha\beta,-\frac{Q^2}{2}}(Q,\varphi) = \sum_{\substack{d>0 \\ Q/d\in\Lambda_{21,5}}} d^2 c_k\big(-\tfrac{Q^2}{2d^2}\big) G^{(20,4)}_{\alpha\beta,-\frac{Q^2}{2d^2}}\big(\tfrac{Q}{d}\big)\,, \tag{6.38}$$

---

[27]Note that CHL models in $D = 3$ all decompactify to the same model in $D = 6$, whose rank is fixed by the constraints of anomaly cancellation.

where $\bar{G}^{(20,4)}_{\alpha\beta} = \bar{G}^{(20,4)}_{F,\alpha\beta} + \bar{G}^{(20,4)}_{P,\alpha\beta}$ as described in §4. The last line corresponds to instanton anti-instanton corrections that are missed by the unfolding method, and which could be computed by solving (E.51) for $Q = 0$.

# Acknowledgements

We are grateful to Massimo Bianchi, Eric d'Hoker, Axel Kleinschmidt, Sameer Murthy, Augusto Sagnotti, Roberto Volpato, for valuable discussions. The work of G.B. is partially supported by the Agence Nationale de la Recherche (ANR) under the grant Black-dS-String (ANR-16-CE31-0004). The research of BP is supported in part by French state funds managed by ANR in the context of the LABEX ILP (ANR-11-IDEX-0004-02, ANR-10- LABX-63).

# A Compendium on Siegel modular forms

## A.1 Action on $\mathcal{H}_2$

The Siegel's upper half plane $\mathcal{H}_2$ is the space of complex symmetric matrices

$$\Omega = \begin{pmatrix} \rho & v \\ v & \sigma \end{pmatrix} \quad \text{such that} \quad |\Omega_2| > 0, \quad \rho_2 > 0, \quad \sigma_2 > 0, \tag{A.1}$$

where $\Omega_1$ and $\Omega_2$ denote the real and imaginary parts of $\Omega$, similarly for $\rho, v, \sigma$, and $|\Omega_2|$ is the determinant of $\Omega_2$. An element $\gamma \in Sp(4, \mathbb{Z})$,

$$\gamma = \begin{pmatrix} A & B \\ C & D \end{pmatrix}, \quad \gamma \varepsilon \gamma^t = \varepsilon, \quad \varepsilon = \begin{pmatrix} 0 & \mathbf{1}_2 \\ -\mathbf{1}_2 & 0 \end{pmatrix}, \tag{A.2}$$

with

$$A^\intercal C - C^\intercal A = 0, \qquad B^\intercal D - D^\intercal B = 0, \qquad A^\intercal D - C^\intercal B = \mathbf{1}_2, \tag{A.3}$$

acts on $\mathcal{H}_2$ via

$$\Omega \mapsto \tilde{\Omega} = (A\Omega + B)(C\Omega + D)^{-1}. \tag{A.4}$$

A standard fundamental domain for the action of $Sp(4, \mathbb{Z})$ on $\mathcal{H}_2$ is the domain $\mathcal{F}_2$ defined by the conditions [87]

$$-\frac{1}{2} < \rho_1, \sigma_1, v_1 < \frac{1}{2}, \quad 0 < 2v_2 \le \rho_2 \le \sigma_2, \quad |C\Omega + D| \ge 1 \tag{A.5}$$

for all $\gamma \in Sp(4, \mathbb{Z})$ (the latter condition needs only to be checked for a finite number of $\gamma$'s).

The period matrix of a genus-two curve $\Sigma$ takes values in $\mathcal{H}_2 \backslash S$, where $S$ is the union of the quadratic divisors

$$D(m_i, j, n_i; \Omega) \equiv m^2 - m^1\rho + n_1\sigma + n_2(\rho\sigma - v^2) + jv = 0, \tag{A.6}$$

parametrized by five integers $M = (m^1, m^2, j, n_1, n_2)$. $M$ transform as a vector under $Sp(4) \sim O(3, 2)$ such that the signature (2,3) quadratic form

$$\Delta(M) = j^2 + 4(m^1 n_1 + m^2 n_2) \tag{A.7}$$

and the parity of $j$ stay invariant. Under a combined action of $\gamma$ on $\Omega$ and $M$, the divisor $D(M; \Omega) = 0$ stays invariant,

$$D(\tilde{M}; \tilde{\Omega}) = [\det(C\Omega + D)]^{-1} D(M, \Omega). \tag{A.8}$$

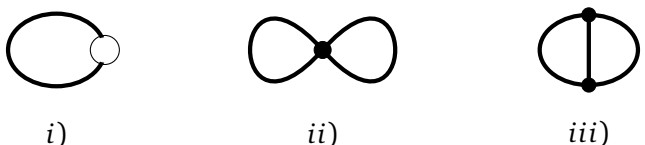

Figure 1: Degenerations of a genus-two Riemann surface corresponding to the boundary strata of the fundamental domain $\mathcal{F}_2$. The white node in i) corresponds to a torus while the black dots in ii), iii) corresponds to a sphere. The 'figure-eight' and 'sunset' diagrams in supergravity are obtained by replacing the black dots in ii) and iii) with supergravity 4-point and 3-point interactions, and attaching four external gauge bosons to the edges.

The divisor $S$ is the locus where the curve $\Sigma$ degenerates into the connected sum of two genus-one curves. Its intersection with the fundamental domain $\mathcal{F}_2$ is simply the divisor $v = 0$.

On the other hand, the boundary of the domain $\mathcal{F}_2$ consists of three strata, i) $\sigma_2 \to +\infty$ where $\Sigma$ degenerates into a one-loop graph, ii) $\rho_2, \sigma_2 \to +\infty$ at the same rate where $\Sigma$ degenerates into a figure-eight graph, and iii) $v_2, \rho_2, \sigma_2 \to +\infty$ at the same rate where $\Sigma$ degenerates into a sunset diagram (see Figure 1). In order to discuss these limits, it will be useful to introduce the alternative parametrizations for $\Omega_2$,

$$\Omega_2 = \begin{pmatrix} \rho_2 & \rho_2 u_2 \\ \rho_2 u_2 & t + \rho_2 u_2^2 \end{pmatrix} = \frac{1}{V \tau_2} \begin{pmatrix} |\tau|^2 & -\tau_1 \\ -\tau_1 & 1 \end{pmatrix} \tag{A.9}$$

such that the limits i) and iii) correspond to $t \to +\infty$ and $V \to 0$, respectively.

We now give the explicit action of some relevant subgroups of $Sp(4, \mathbb{Z})$:

1. $SL(2)_\rho$ (leaving $t = \sigma_2 - v_2^2/\rho_2$ invariant)

$$\begin{pmatrix} a & b \\ c & d \end{pmatrix}_\rho = \begin{pmatrix} a & 0 & b & 0 \\ 0 & 1 & 0 & 0 \\ c & 0 & d & 0 \\ 0 & 0 & 0 & 1 \end{pmatrix} : \quad (\rho, v, \sigma)' = \left( \frac{a\rho + b}{c\rho + d}, \frac{v}{c\rho + d}, \sigma - \frac{cv^2}{c\rho + d} \right), \tag{A.10}$$

$$(m^1, m^2, j, n_1, n_2)' = \left( dm^1 + cm^2, bm^1 + am^2, j, an_1 - bn_2, dn_2 - cn_1 \right).$$

We denote by $S_\rho$ the generator $\begin{pmatrix} 0 & -1 \\ 1 & 0 \end{pmatrix}_\rho$.

2. $SL(2)_\sigma$ (leaving $t' = \rho_2 - v_2^2/\sigma_2$ invariant):

$$\begin{pmatrix} a & b \\ c & d \end{pmatrix}_\sigma = \begin{pmatrix} 1 & 0 & 0 & 0 \\ 0 & a & 0 & b \\ 0 & 0 & 1 & 0 \\ 0 & c & 0 & d \end{pmatrix} : \quad (\rho, v, \sigma)' = \left( \rho - \frac{cv^2}{c\sigma + d}, \frac{v}{c\sigma + d}, \frac{a\sigma + b}{c\sigma + d} \right), \tag{A.11}$$

$$(m^1, m^2, j, n_1, n_2)' = \left( am^1 + bn_2, am^2 - bn_1, j, dn_1 - cm^2, cm^1 + dn_2 \right).$$

We denote by $S_\sigma$ the generator $\begin{pmatrix} 0 & -1 \\ 1 & 0 \end{pmatrix}_\sigma$.

3. $\text{Heis}_\rho$ (leaving $\Omega_2$ invariant):

$$T_{\lambda,\mu,\kappa} = \begin{pmatrix} 1 & 0 & 0 & \mu \\ \lambda & 1 & \mu & \kappa \\ 0 & 0 & 1 & -\lambda \\ 0 & 0 & 0 & 1 \end{pmatrix} :$$

$$(\rho, v, \sigma)' = \left(\rho, \mu + \lambda\rho + v, \sigma + \kappa + 2\lambda v + \lambda\mu + \lambda^2\rho\right),$$

$$(m^1, m^2)' = \left(m^1 + j\lambda + (\mu n_2 - \lambda n_1)\lambda + \kappa n_2, m^2 - \mu(j - \lambda n_1 + \mu n_2) - \kappa n_1\right),$$

$$(j, n_1, n_2)' = (j - 2\lambda n_1 + 2\mu n_2, n_1, n_2).$$

(A.12)

4. $\text{Heis}_\sigma$ (leaving $\Omega_2$ invariant):

$$\tilde{T}_{\lambda,\mu,\kappa} = \begin{pmatrix} 1 & \lambda & \kappa & \mu \\ 0 & 1 & \mu & 0 \\ 0 & 0 & 1 & 0 \\ 0 & 0 & -\lambda & 1 \end{pmatrix}.$$

(A.13)

5. $GL(2, \mathbb{Z})_S$ (leaving $V = 1/\sqrt{|\Omega_2|}$ invariant):

$$\begin{pmatrix} a & b \\ c & d \end{pmatrix}_S = \begin{pmatrix} a & -b & 0 & 0 \\ -c & d & 0 & 0 \\ 0 & 0 & d & c \\ 0 & 0 & b & a \end{pmatrix} :$$

$$(\rho, v, \sigma)' = \left(a^2\rho - 2ab\, v + b^2\sigma, -ac\rho + (ad + bc)v - bd\,\sigma, c^2\rho - 2cd\, v + d^2\,\sigma\right),$$

$$(m^1, m^2)' = \left(-c^2\, n_1 - cd\, j + d^2\, m^1, m^2\right),$$

$$(j, n_1, n_2)' = \left(j + 2bc j - 2bd\, m^1 + 2ac\, n_1, a^2\, n_1 + ab\, j - b^2\, m^1, n_2\right),$$

(A.14)

$$\tau \mapsto \frac{a\tau + b}{c\tau + d} \quad (ad - bc = 1), \qquad \tau \mapsto \frac{a\bar{\tau} + b}{c\bar{\tau} + d} \quad (ad - bc = -1).$$

(A.15)

Defining $\Omega_2 = \begin{pmatrix} L_1 + L_2 & L_2 \\ L_2 & L_2 + L_3 \end{pmatrix}$, the permutations of the $L_i$'s correspond to the following elements of $GL(2, \mathbb{Z})_S$:

$$L_1 \leftrightarrow L_2 : \begin{pmatrix} 0 & 1 \\ 1 & 0 \end{pmatrix}_S, \quad L_2 \leftrightarrow L_3 : \begin{pmatrix} 1 & 1 \\ 0 & -1 \end{pmatrix}_S, \quad L_1 \leftrightarrow L_3 : \begin{pmatrix} 1 & 0 \\ -1 & -1 \end{pmatrix}_S.$$

(A.16)

6. $\mathbb{Z}^3$ (leaving $\Omega_2$ invariant):

$$T_{r_1, r_2, r_3} = \begin{pmatrix} 1 & 0 & r_1 & r_2 \\ 0 & 1 & r_2 & r_3 \\ 0 & 0 & 1 & 0 \\ 0 & 0 & 0 & 1 \end{pmatrix} : \quad (\rho, v, \sigma)' = (\rho + r_1, v + r_2, \sigma + r_3),$$

$$(m^1, m^2)' = \left(m^1 + n_2 r_3, m^2 - n_2 r_2^2 - j r_2 + m^1 r_1 - n_1 r_3 + n_2 r_1 r_3\right),$$

$$(j, n_1, n_2)' = (j + 2n_2 r_2, n_1 - n_2 r_1, n_2).$$

(A.17)

7. $\sigma_{\rho \leftrightarrow \sigma}$:

$$h_{\rho \leftrightarrow \sigma} = \begin{pmatrix} 0 & 1 & 0 & 0 \\ 1 & 0 & 0 & 0 \\ 0 & 0 & 0 & 1 \\ 0 & 0 & 1 & 0 \end{pmatrix} : \quad (\rho, v, \sigma)' = (\sigma, v, \rho),$$

(A.18)

$$(m^1, m^2, j, n_1, n_2)' = (n_1, -m^2, -j, m_1, -n_2).$$

## A.2 Siegel modular forms and congruence subgroups

For any $\gamma \in Sp(4, \mathbb{R})$ and integer $w$, we define the Petersson slash operator

$$(\Phi|_w \gamma)(\Omega) = [\det(C\Omega + D)]^{-w} \Phi\left((A\Omega + B)(C\Omega + D)^{-1}\right) . \tag{A.19}$$

A Siegel modular form $\Phi(\Omega) = \Phi(\rho, \sigma, \nu)$ of weight $w$ under a subgroup $\Gamma \subset Sp(4, \mathbb{Z})$ satisfies $\Phi|_w \gamma = \Phi$ for any $\gamma \in \Gamma$. We shall be mostly interested in modular forms with respect to the congruence subgroups of $Sp(4, \mathbb{Z})$ (A.2), denoting its elements by $\left(\begin{smallmatrix} A & B \\ C & D \end{smallmatrix}\right)$,

1. $\Gamma_{2,0}(N)$, restricting to elements with $C = 0 \bmod N$;

2. $\tilde{\Gamma}_{2,0}(N) = S_\rho \cdot \Gamma_{2,0}(N) \cdot S_\rho^{-1}$, its conjugate w.r.t. $S_\rho$ (A.10);

3. $\hat{\Gamma}_{2,0}(N) = S_\sigma \cdot \Gamma_{2,0}(N) \cdot S_\sigma^{-1}$, conjugate of $\Gamma_{2,0}(N)$ w.r.t. $S_\sigma$ (A.11);

4. $\Gamma_{2,1}(N) \subset \Gamma_{2,0}(N)$, restricting to elements with $A = D = 1 \bmod N$;

5. $\Gamma_2(N) \subset \Gamma_{2,1}(N)$, restricting to elements with $B = 0 \bmod N$;

6. $\Gamma_{2,e_r}(N)$ the subgroup fixing the vector $(0, 0, 0, r)$ modulo $N$;

7. $\Gamma_{2,0,e_r}(N) = \Gamma_{2,e_r}(N) \cap \Gamma_{2,0}(N)$.

The indices of these subgroups inside $Sp(4, \mathbb{Z})$ are summarized below:

$$
\begin{aligned}
\left| Sp(4, \mathbb{Z})/\Gamma_2(N) \right| &= N^{10} \prod_{p|N} \left(1 - \frac{1}{p^2}\right)\left(1 - \frac{1}{p^4}\right), \\
\left| Sp(4, \mathbb{Z})/\Gamma_{2,0}(N) \right| &= N^3 \prod_{p|N} \left(1 + \frac{1}{p}\right)\left(1 + \frac{1}{p^2}\right), \\
\left| Sp(4, \mathbb{Z})/\Gamma_{2,1}(N) \right| &= N^7 \prod_{p|N} \left(1 - \frac{1}{p^2}\right)\left(1 - \frac{1}{p^4}\right), \\
\left| Sp(4, \mathbb{Z})/\Gamma_{2,0,e_r}(N) \right| &= \frac{N^5}{r^2} \prod_{p|N} \left(1 + \frac{1}{p}\right)\left(1 + \frac{1}{p^2}\right) \prod_{p'|\frac{N}{r}} \left(1 - \frac{1}{p'^2}\right), \\
\left| Sp(4, \mathbb{Z})/\Gamma_{2,e_1}(N) \right| &= N^4 \prod_{p|N} \left(1 - \frac{1}{p^4}\right),
\end{aligned}
\tag{A.20}
$$

where $p, p'$ run over primes. Indeed the corresponding quotients can be understood as

$$
\begin{aligned}
\left| \Gamma_{2,1}(N)/\Gamma_2(N) \right| &= N^3 = \left(\mathbb{Z}/N\mathbb{Z}\right)^3_B, \\
\left| \Gamma_{2,0,e_1}(N)/\Gamma_{2,1}(N) \right| &= N^2 \prod_{p|N} \left(1 - \frac{1}{p}\right) = \left| \Gamma_1(N)/\Gamma(N) \right|_D \times \left| \Gamma_{2,0}(N)/\Gamma_{2,1}(N) \right|_{\left[\begin{smallmatrix} a_1 b_1 \\ c_1 d_1 \end{smallmatrix}\right]}, \\
\left| \Gamma_{2,0,e_r}(N)/\Gamma_{2,0,e_1}(N) \right| &= r^2 \prod_{\substack{p|N \\ p \nmid \frac{N}{r}}} \left(1 - \frac{1}{p^2}\right) = \left| \Gamma_1(\tfrac{N}{r})/\Gamma_1(N) \right|_D, \\
\left| \Gamma_{2,0}(N)/\Gamma_{2,0,e_r}(N) \right| &= (\tfrac{N}{r})^2 \prod_{p|\frac{N}{r}} \left(1 - \frac{1}{p^2}\right) = \left| SL(2, \mathbb{Z})/\Gamma_1(\tfrac{N}{r}) \right|_D,
\end{aligned}
\tag{A.21}
$$

where the subscript indicates the embedding $SL(2, \mathbb{Z}) \subset Sp(4, \mathbb{Z})$ of the coset representatives.

Of special interest is the Hecke congruence subgroup $\Gamma_{2,0}(N)$ and its conjugates $\tilde{\Gamma}_{2,0}(N)$, $\hat{\Gamma}_{2,0}(N)$. The cosets of $Sp(4, \mathbb{Z})/\Gamma_{2,0}(N)$ are in one-to-one correspondence with cosets of

$GSp_N(4,\mathbb{Z})/Sp(4,\mathbb{Z})$, where $GSp_N(4,\mathbb{Z})$ is the group of symplectic similitudes such that $\gamma \varepsilon \gamma^t = N\varepsilon$. For $N$ prime, the $(N+1)(N^2+1) = 1 + N + N^2 + N^3$ cosets can be chosen as (see e.g. [88, p.6])

$$
\begin{pmatrix} N & & & \\ & N & & \\ & & 1 & \\ & & & 1 \end{pmatrix}, \quad
\begin{pmatrix} 1 & & a & \\ & N & & \\ & & N & \\ & & & 1 \end{pmatrix}, \quad
\begin{pmatrix} N & & & \\ -a & 1 & & b \\ & & 1 & a \\ & & & N \end{pmatrix}, \quad
\begin{pmatrix} 1 & & a & c \\ & 1 & c & b \\ & & N & \\ & & & N \end{pmatrix}, \quad \text{(A.22)}
$$

with $a, b, c = 0 \dots N-1$. For $\Phi(\rho, \sigma, \nu)$ a Siegel modular form of weight $w$ for the full Siegel modular group $Sp(4,\mathbb{Z})$, the sum of the action of these elements on $\Phi$ produces again a Siegel modular form for the full Siegel modular group $Sp(4,\mathbb{Z})$, which is the image of $\Phi$ under the $N$-th Hecke operator $H_N$,

$$
\begin{aligned}
H_N \Phi(\rho, \sigma, \nu) = {} & \Phi(N\rho, N\sigma, N\nu) + N^{-w} \sum_{a \bmod N} \Phi\left(\frac{\rho+a}{N}, N\sigma, \nu\right) \\
& + N^{-w} \sum_{a,b \bmod N} \Phi\left(N\rho, \frac{\sigma - 2a\nu + a^2\rho + b}{N}, \nu - a\rho\right) + N^{-2w} \sum_{a,b,c \bmod N} \Phi\left(\frac{\rho+a}{N}, \frac{\sigma+b}{N}, \frac{\nu+c}{N}\right).
\end{aligned}
$$
(A.23)

The first term in this sum, $\Phi(N\rho, N\sigma, N\nu)$, is then a Siegel modular form for $\Gamma_{2,0}(N)$. The 'Fricke involution'

$$
\Phi \mapsto \Phi|_w \begin{pmatrix} 0 & 0 & 0 & \frac{1}{\sqrt{N}} \\ 0 & 0 & -\frac{1}{\sqrt{N}} & 0 \\ 0 & -\sqrt{N} & 0 & 0 \\ \sqrt{N} & 0 & 0 & 0 \end{pmatrix} = \Phi|_w \begin{pmatrix} 0 & 0 & -\frac{1}{\sqrt{N}} & 0 \\ 0 & 0 & 0 & -\frac{1}{\sqrt{N}} \\ \sqrt{N} & 0 & 0 & 0 \\ 0 & \sqrt{N} & 0 & 0 \end{pmatrix} = [N|\Omega|]^{-w} \Phi\left(-(N\Omega)^{-1}\right) \quad \text{(A.24)}
$$

takes a Siegel modular form $\Phi$ of weight $w$ under $\Gamma_{2,0}(N)$ into another one. Similarly,

$$
\widetilde{\Phi} \mapsto \widetilde{\Phi}|_w \begin{pmatrix} 0 & 1/\sqrt{N} & 0 & 0 \\ \sqrt{N} & 0 & 0 & 0 \\ 0 & 0 & 0 & \sqrt{N} \\ 0 & 0 & 1/\sqrt{N} & 0 \end{pmatrix} = \widetilde{\Phi}(\sigma/N, N\rho, \nu) \quad \text{(A.25)}
$$

takes a Siegel modular form $\tilde{\Phi}$ of weight $w$ under $\widetilde{\Gamma}_{2,0}(N)$ into another one.

## A.3   Genus two theta series

The genus-two even theta series are defined as

$$
\vartheta^{(2)}\left[{}^{a_1,a_2}_{b_1,b_2}\right](\Omega, \zeta) = \sum_{p_1, p_2 \in \mathbb{Z}} e^{i\pi \left(p_1 + \frac{a_1}{2}, p_2 + \frac{a_2}{2}\right)^t \Omega \begin{pmatrix} p_1 + \frac{a_1}{2} \\ p_2 + \frac{a_2}{2} \end{pmatrix} + 2\pi i \left(p_1 + \frac{a_1}{2}, p_2 + \frac{a_2}{2}\right)^t \begin{pmatrix} \zeta_1 + \frac{b_1}{2} \\ \zeta_2 + \frac{b_2}{2} \end{pmatrix}}, \quad \text{(A.26)}
$$

with $a_i, b_i \in \mathbb{Z}$. It is an even or odd function of $\zeta = (\zeta_1, \zeta_2)^t$ depending on the parity of $a_1 b_1 + a_2 b_2$. When it is even, the value at $\zeta = 0$ is the Thetanullwert denoted by $\vartheta^{(2)}\left[{}^{a_1,a_2}_{b_1,b_2}\right](\Omega)$. The value of $a_i, b_i$ modulo two defines a spin structure labelled by the column vector $\kappa = (a_1, a_2, b_1, b_2)^t$, whose parity is that of $a_1 b_1 + a_2 b_2$. Under translations of the characteristics by even integers,

$$
\vartheta^{(2)}\left[{}^{a_1+2a_1',a_2+2a_2'}_{b_1+2b_1',b_2+2b_2'}\right](\Omega, \zeta) = e^{i\pi(a_1 b_1' + a_2 b_2')} \vartheta^{(2)}\left[{}^{a_1,a_2}_{b_1,b_2}\right](\Omega, \zeta). \quad \text{(A.27)}
$$

Under $Sp(4,\mathbb{Z})$ transformations,

$$
\vartheta^{(2)}[\tilde{\kappa}](\tilde{\Omega}, \tilde{\zeta}) = \epsilon(\kappa, \gamma) \left[\det(C\Omega + D)\right]^{1/2} \vartheta^{(2)}[\kappa](\Omega, \zeta), \quad \text{(A.28)}
$$

with $\tilde{\Omega} = (A\Omega + B)(C\Omega + D)^{-1}$, $\tilde{\zeta} = (C\Omega + D)^{-t}\zeta$,

$$\tilde{\kappa} = \begin{pmatrix} D & -C \\ -B & A \end{pmatrix}\kappa + \frac{1}{2}\text{diag}\begin{pmatrix} CD^t \\ AB^t \end{pmatrix} \bmod 2 \tag{A.29}$$

and $\epsilon(\kappa, \gamma)$ is an 8-th root of unity. In particular,

$$
\begin{aligned}
\vartheta^{(2)}\big[\begin{smallmatrix} a_1, a_2 \\ b_1, b_2 \end{smallmatrix}\big](\rho + 1, \sigma, v) &= e^{-\frac{i\pi}{4}a_1(a_1+2)}\vartheta^{(2)}\big[\begin{smallmatrix} a_1, a_2 \\ a_1+b_1+1, b_2 \end{smallmatrix}\big](\rho, \sigma, v), \\
\vartheta^{(2)}\big[\begin{smallmatrix} a_1, a_2 \\ b_1, b_2 \end{smallmatrix}\big](\rho, \sigma + 1, v) &= e^{-\frac{i\pi}{4}a_2(a_2+2)}\vartheta^{(2)}\big[\begin{smallmatrix} a_1, a_2 \\ b_1, a_2+b_2+1 \end{smallmatrix}\big](\rho, \sigma, v), \\
\vartheta^{(2)}\big[\begin{smallmatrix} a_1, a_2 \\ b_1, b_2 \end{smallmatrix}\big](\rho, \sigma, v+1) &= e^{-\frac{i\pi}{2}a_1 a_2}\vartheta^{(2)}\big[\begin{smallmatrix} a_1, a_2 \\ b_1+a_2, b_2+a_1 \end{smallmatrix}\big](\rho, \sigma, v), \\
\vartheta^{(2)}\big[\begin{smallmatrix} a_1, a_2 \\ b_1, b_2 \end{smallmatrix}\big](\rho, \sigma + \rho - 2v, v - \rho) &= \vartheta^{(2)}\big[\begin{smallmatrix} a_1-a_2, a_2 \\ b_1, b_1+b_2 \end{smallmatrix}\big](\rho, \sigma, v), \\
\vartheta^{(2)}\big[\begin{smallmatrix} a_1, a_2 \\ b_1, b_2 \end{smallmatrix}\big](-1/\rho, \sigma - v^2/\rho, v/\rho) &= \sqrt{-i\rho}\, e^{\frac{i\pi}{2}a_1 b_1}\vartheta^{(2)}\big[\begin{smallmatrix} b_1, a_2 \\ -a_1, b_2 \end{smallmatrix}\big](\rho, \sigma, v), \\
\vartheta^{(2)}\big[\begin{smallmatrix} a_1, a_2 \\ b_1, b_2 \end{smallmatrix}\big](\rho - v^2/\rho, -1/\sigma, v/\sigma) &= \sqrt{-i\sigma}\, e^{\frac{i\pi}{2}a_2 b_2}\vartheta^{(2)}\big[\begin{smallmatrix} a_1, b_2 \\ b_1, -a_2 \end{smallmatrix}\big](\rho, \sigma, v).
\end{aligned}
\tag{A.30}
$$

In the separating degeneration limit,

$$\vartheta^{(2)}\big[\begin{smallmatrix} a_1 a_2 \\ b_1 b_2 \end{smallmatrix}\big] \overset{v \to 0}{\to} \begin{cases} \vartheta\big[\begin{smallmatrix} a_1 \\ b_1 \end{smallmatrix}\big](\rho)\vartheta\big[\begin{smallmatrix} a_2 \\ b_2 \end{smallmatrix}\big](\sigma) & \big[\begin{smallmatrix} a_1 a_2 \\ b_1 b_2 \end{smallmatrix}\big] \neq \big[\begin{smallmatrix} 11 \\ 11 \end{smallmatrix}\big] \\ \frac{v}{2\pi i}\vartheta\big[\begin{smallmatrix} 1 \\ 1 \end{smallmatrix}\big]'(\rho)\vartheta\big[\begin{smallmatrix} 1 \\ 1 \end{smallmatrix}\big]'(\sigma) & \big[\begin{smallmatrix} a_1 a_2 \\ b_1 b_2 \end{smallmatrix}\big] = \big[\begin{smallmatrix} 11 \\ 11 \end{smallmatrix}\big] \end{cases}, \tag{A.31}$$

where $\vartheta\big[\begin{smallmatrix} a \\ b \end{smallmatrix}\big]$ is the genus-one theta series,

$$\vartheta\big[\begin{smallmatrix} a \\ b \end{smallmatrix}\big] = \sum_{p \in \mathbb{Z}} e^{i\pi(p+\frac{a}{2})^2 \tau + i\pi b(p+\frac{a}{2})} \tag{A.32}$$

and $\vartheta\big[\begin{smallmatrix} 1 \\ 1 \end{smallmatrix}\big]'(\rho) = 2\pi\eta^3$, $\vartheta_2\vartheta_3\vartheta_4 = 2\eta^3$.

## A.4 Meromorphic Siegel modular forms from Borcherds products

In the context of heterotic CHL orbifolds, two meromorphic Siegel modular forms $\Phi_{k-2}$ and $\tilde{\Phi}_{k-2}$ of weight $k-2$ under $\Gamma_{2,0}(N)$ and $\tilde{\Gamma}_{2,0}(N)$, respectively play an essential rôle. They are given by infinite products [89] [90, 3.16,3.17] [29, C.18,C.19][28]

$$\Phi_{k-2}(\rho, \sigma, v) = e^{2\pi i(\rho+\sigma+v)}\prod_{r=0}^{N-1}\prod_{\substack{k', \ell, j \in \mathbb{Z} \\ k', \ell \geq 0, \\ j < 0 \text{ for } k'=\ell=0}}\left(1 - e^{2\pi i r/N}e^{2\pi i(k'\sigma + \ell\rho + jv)}\right)^{\sum_{s=0}^{N-1}e^{-\frac{2\pi i rs}{N}}c_{j\bmod 2}^{(0,s)}(4k'\ell - j^2)}, \tag{A.33}$$

$$\tilde{\Phi}_{k-2}(\rho, \sigma, v) = e^{2\pi i(\sigma + \frac{1}{N}\rho + v)}\prod_{r=0}^{N-1}\prod_{\substack{k' \in \mathbb{Z} + \frac{r}{N}, \ell, j \in \mathbb{Z} \\ k', \ell \geq 0, \\ j < 0 \text{ for } k'=\ell=0}}\left(1 - e^{2\pi i(k'\rho + \ell\sigma + jv)}\right)^{\sum_{s=0}^{N-1}e^{-\frac{2\pi i s\ell}{N}}c_{j\bmod 2}^{(r,s)}(4k'\ell - j^2)}. \tag{A.34}$$

Here, $c_b^{(r,s)}(n)$ with $b \in \mathbb{Z}/(2\mathbb{Z})$ are Fourier coefficients of a family of index 1 weak Jacobi forms

$$F^{(r,s)}(\tau, z) = \sum_{j \in \mathbb{Z}, n \in \frac{\mathbb{Z}}{N}} c_{j\bmod 2}^{(r,s)}(4n - j^2)e^{2\pi i(n\tau + jz)} \tag{A.35}$$

---

[28]Note that $\Phi_{k-2}(\rho, \sigma, v)$ and $\tilde{\Phi}_{k-2}(\rho, \sigma, v)$ are denoted by $\widehat{\Phi}(\rho, \sigma, v)$ and $\tilde{\Phi}(\sigma, \rho, v)$ in [29], while $\tilde{\Phi}(\rho, \sigma, v)$ coincides with $\Phi_{g,e}(\rho, \sigma, v)$ in [30].

obtained as a twining/twisted elliptic genus of the $\mathbb{Z}_N$ orbifold of $K3$. In particular, for $N = 1, 2, 3, 5, 7$ and $1 \leq s \leq N - 1$,

$$F^{(0,0)} = \frac{2}{N}\phi_{0,1}\,,\quad F^{(0,s)} = \frac{2}{N(N+1)}\phi_{0,1} + \frac{2(E_2(\tau)-NE_2(N\tau))}{(N+1)(N-1)}\phi_{-2,1}\,,$$

$$F^{(r,s)} = \frac{2}{N(N+1)}\phi_{0,1} - \frac{2(E_2(\frac{\tau+s/r}{N})-NE_2(\tau))}{N(N+1)(N-1)}\phi_{-2,1}\,,$$

(A.36)

where $\phi_{0,1} = 4\sum_{i=2,3,4}\left(\frac{\vartheta_i(\tau,z)}{\vartheta_i(\tau,0)}\right)^2$, $\phi_{-2,1} = \vartheta_1^2(\tau,z)/\eta^6$ are the standard generators of the ring of weak Jacobi forms, and $s/r = sk$ where $kr = 1 \bmod N$. It is also useful to consider the discrete Fourier transform of the coefficients $c_b^{(r,s)}(n)$ with respect to $s$,

$$\hat{c}_b^{(r,s)}(n) = \sum_{s'=0}^{N-1} e^{-2\pi i s s'/N} c_b^{(r,s')}(n)\,. \tag{A.37}$$

Using the property $\hat{c}_j^{(r,s)}(n) = \hat{c}_j^{(s,r)}(n)$, one can rewrite $\tilde{\Phi}_{k-2}$ as [30, 5.10]

$$\tilde{\Phi}_{k-2}(\rho,\sigma,\nu) = e^{2\pi i(\sigma+\frac{\rho}{N}+\nu)}\prod_{\substack{k'\in\mathbb{Z}/N,\ell,j\in\mathbb{Z}\\k',\ell\geq 0, j<0 \text{ for } k'=\ell=0}}\left(1 - e^{2\pi i(k'\rho+\ell\sigma+j\nu)}\right)^{\sum_{s=0}^{N-1} e^{-2\pi i s k'} c_{j\bmod 2}^{(\ell,s)}(4k'\ell-j^2)}\,.$$

(A.38)

From this relation, it is manifest that $\tilde{\Phi}_{k-2}$ is invariant under the Fricke involution [89, §C],

$$\tilde{\Phi}_{k-2}(\rho,\sigma,\nu) = \tilde{\Phi}_{k-2}(N\sigma,\rho/N,\nu) = \tilde{\Phi}_{k-2}\bigg|\begin{pmatrix} 0 & \sqrt{N} & 0 & 0 \\ 1/\sqrt{N} & 0 & 0 & 0 \\ 0 & 0 & 0 & 1/\sqrt{N} \\ 0 & 0 & \sqrt{N} & 0 \end{pmatrix}\,, \tag{A.39}$$

and therefore, so is $\Phi_{k-2}$,

$$\Phi_{k-2}(\Omega) = (N|\Omega|)^{2-k}\Phi_{k-2}(-1/(N\Omega)) = \Phi_{k-2}\bigg|\begin{pmatrix} 0 & 0 & 0 & 1/\sqrt{N} \\ 0 & 0 & -1/\sqrt{N} & 0 \\ 0 & -\sqrt{N} & 0 & 0 \\ \sqrt{N} & 0 & 0 & 0 \end{pmatrix}\,. \tag{A.40}$$

It is worth recalling that the infinite products (A.33) and (A.34) arise as theta liftings of $F^{(r,s)}$, namely

$$\text{R.N.} \int_{\mathcal{F}_1} d\mu_1 \sum_{\substack{m_1,n_1,j\in\mathbb{Z}\\m_2\in\mathbb{Z}/N,n_2\in N\mathbb{Z}+r}} q^{\frac{1}{2}p_L^2}\bar{q}^{\frac{1}{2}p_R^2} e^{2\pi i m_2 s} h_{j\bmod 2}^{(r,s)} = -2\log|\Omega_2|^{2(k-2)}|\Phi_{k-2}(\rho,\sigma,\nu)|^2\,,$$

$$\text{R.N.} \int_{\mathcal{F}_1} d\mu_1 \sum_{\substack{m_1,m_2,n_2,j\in\mathbb{Z}\\n_1\in\mathbb{Z}+\frac{r}{N}}} q^{\frac{1}{2}p_L^2}\bar{q}^{\frac{1}{2}p_R^2} e^{2\pi i m_1 s/N} h_{j\bmod 2}^{(r,s)} = -2\log|\Omega_2|^{2(k-2)}|\tilde{\Phi}_{k-2}(\sigma,\rho,\nu)|^2\,,$$

(A.41)

where $h_b^{(r,s)}$, $b \in \mathbb{Z}/(2\mathbb{Z})$ is the vector valued modular form arising in the theta series decomposition

$$F^{(r,s)}(\tau,z) = h_0^{(r,s)}(\tau)\vartheta_3(2\tau,2z) + h_1^{(r,s)}(\tau)\vartheta_2(2\tau,2z)\,,\qquad h_b^{(r,s)} = \sum_{n\in\frac{1}{N}\mathbb{Z}-\frac{b^2}{4}} c_b^{(r,s)}(4n)e^{2\pi i n\tau}$$

(A.42)

and $p_R$, $p_L$ are projections of the vector $M = (m_1, n_1, j, m^2, n^2)$ such that

$$p_R^2 = \frac{1}{2|\Omega_2|} \left| m^2 - m^1\rho + n_1\sigma + n_2(\rho\sigma - v^2) + jv \right|^2 , \quad \frac{1}{2}(p_L^2 - p_R^2) = m^1 n_1 + m^2 n_2 + \frac{j^2}{4} .$$
(A.43)

From the infinite product representation, one can easily read off the location of the zeros and poles which intersect the cusp $\Omega_2 = i\infty$. Such zeros (respectively, poles) arise from the existence of positive (respectively, negative) coefficients $\hat{c}^{(r,s)}(m)$ with $m < 0$, known as polar coefficients. For $N = 1, 2, 3, 5, 7$, the only positive polar term is $\hat{c}_1^{(0,0)}(-1) = 2$, which implies that $\Phi_{k-2}$ and $\tilde{\Phi}_{k-2}$ have a double zero on the diagonal locus $v = 0$, where they behave according to [29]

$$\begin{aligned}
\Phi_{k-2}(\rho, \sigma, v) &\sim -4\pi^2 v^2 \Delta_k(\rho)\Delta_k(\sigma) , \\
\tilde{\Phi}_{k-2}(\rho, \sigma, v) &\sim -4\pi^2 v^2 \Delta_k(\rho/N)\Delta_k(\sigma) ,
\end{aligned}$$
(A.44)

where $\Delta_k(\rho) = \eta^k(\rho)\eta^k(N\rho)$. It can be shown that all zeros of $\Phi_{k-2}$ and $\tilde{\Phi}_{k-2}$ occur only on the divisor $v = 0$ and its images under the congruence subgroups $\Gamma_{2,0}(N)$ and $\tilde{\Gamma}_{2,0}(N)$, respectively. For $N = 1, 2, 3$, $\hat{c}_1^{(0,0)}(-1)$ is the only polar term, so $\Phi_{k-2}$ and $\tilde{\Phi}_{k-2}$ are actually holomorphic Siegel modular forms, corresponding to the Igusa cusp form $\Phi_{10}$ for $N = 1$, or the cusp forms $\Phi_6$ of level 2 and $\Phi_4$ of level 3 constructed in [91, 92]. In particular, $\Phi_{10}$ is proportional to the product of the square of the ten even Thetanullwerte,

$$\Phi_{10} = 2^{-20} \left( \vartheta^{(2)}\begin{bmatrix} 00 \\ 00 \end{bmatrix} \vartheta^{(2)}\begin{bmatrix} 01 \\ 00 \end{bmatrix} \vartheta^{(2)}\begin{bmatrix} 10 \\ 00 \end{bmatrix} \vartheta^{(2)}\begin{bmatrix} 00 \\ 10 \end{bmatrix} \vartheta^{(2)}\begin{bmatrix} 00 \\ 01 \end{bmatrix} \vartheta^{(2)}\begin{bmatrix} 01 \\ 10 \end{bmatrix} \vartheta^{(2)}\begin{bmatrix} 10 \\ 01 \end{bmatrix} \vartheta^{(2)}\begin{bmatrix} 11 \\ 00 \end{bmatrix} \vartheta^{(2)}\begin{bmatrix} 00 \\ 11 \end{bmatrix} \vartheta^{(2)}\begin{bmatrix} 11 \\ 11 \end{bmatrix} \right)^2 ,$$
(A.45)

while $\Phi_6$ is proportional to the product of the square of 6 among the ten even Thetanullwerte,

$$\Phi_6 = 2^{-12} \left( \vartheta^{(2)}\begin{bmatrix} 01 \\ 00 \end{bmatrix} \vartheta^{(2)}\begin{bmatrix} 01 \\ 10 \end{bmatrix} \vartheta^{(2)}\begin{bmatrix} 10 \\ 00 \end{bmatrix} \vartheta^{(2)}\begin{bmatrix} 10 \\ 01 \end{bmatrix} \vartheta^{(2)}\begin{bmatrix} 11 \\ 00 \end{bmatrix} \vartheta^{(2)}\begin{bmatrix} 11 \\ 11 \end{bmatrix} \right)^2 .$$
(A.46)

For $N = 5$ and $N = 7$, there are additional polar coefficients but they are all negative, implying that $\Phi_{k-2}$ and $\tilde{\Phi}_{k-2}$ have poles,

$$\begin{aligned}
N = 5 &: \hat{c}_1^{(1,1)}(-\tfrac{1}{5}) = \hat{c}_1^{(2,3)}(-\tfrac{1}{5}) = \hat{c}_1^{(3,2)}(-\tfrac{1}{5}) = \hat{c}_1^{(4,4)}(-\tfrac{1}{5}) = -2 \\
N = 7 &: \hat{c}_1^{(1,1)}(-\tfrac{3}{7}) = \hat{c}_1^{(2,4)}(-\tfrac{3}{7}) = \hat{c}_1^{(3,5)}(-\tfrac{3}{7})) = \hat{c}_1^{(4,2)}(\tfrac{3}{7})) = \hat{c}_1^{(5,3)}(\tfrac{3}{7})) = \hat{c}_1^{(6,6)}(\tfrac{3}{7}) = -1 .
\end{aligned}$$
(A.47)

Note however that the Siegel modular forms relevant for our problem are the inverse of $\Phi_{k-2}$ and $\tilde{\Phi}_{k-2}$, which have a double pole on the diagonal locus $v = 0$ for all $N$.

From the infinite product representation one can also read-off the behavior of $1/\Phi_{k-2}$ and $1/\tilde{\Phi}_{k-2}$ in the maximal non-separating degeneration $\Omega_2 \to \infty$, obtained by setting

---

[29]Note that these two equations are consistent with (2.10) since $\Delta_k$ is invariant under the Fricke involution, *i.e.* $\Delta_k(-1/\rho) = (i\sqrt{N})^{-k} \rho^k \Delta_k(\rho/N)$

$e^{2\pi i\rho} = q_1 q_3, e^{2\pi i\sigma} = q_2 q_3, e^{2\pi i v} = q_3$, and Taylor expanding near $q_i \to 0$:

$$\frac{1}{\Phi_{10}} = \frac{1}{q_1 q_2 q_3} + 2\sum_{i<j}\frac{1}{q_i q_j} + \left[24\sum_{i=1}^{3}\frac{1}{q_i} + 3\sum_{i\neq j<k\neq i}\frac{q_i}{q_j q_k}\right]$$

$$+ \left[0 + 48\sum_{i\neq j}\frac{q_i}{q_j} + 4\sum_{i\neq j<k\neq i}\frac{q_i^2}{q_j q_k}\right] + \mathcal{O}(q_i), \tag{A.48}$$

$$\frac{1}{\Phi_{k-2}} = \frac{1}{q_1 q_2 q_3} + 2\sum_{i<j}\frac{1}{q_i q_j} + \left[\frac{24}{N+1}\sum_{i=1}^{3}\frac{1}{q_i} + 3\sum_{i\neq j<k\neq i}\frac{q_i}{q_j q_k}\right]$$

$$+ \left[\frac{48N}{N^2-1} + \frac{48}{N+1}\sum_{i\neq j}\frac{q_i}{q_j} + 4\sum_{i\neq j<k\neq i}\frac{q_i^2}{q_j q_k}\right] + \mathcal{O}(q_i), \tag{A.49}$$

$$\frac{1}{\tilde{\Phi}_{k-2}} = \frac{1}{q_1^{1/N} q_2 q_3^{1/N}} + \frac{24}{N+1}\frac{1}{q_2} - \frac{48}{N^2-1} + \dots, \tag{A.50}$$

where the dot denotes terms involving positive powers of $q_i$. Since $Sp(4,\mathbb{Z})$ and its congruence subgroup $\Gamma_{2,0}(N)$ contains $GL(2,\mathbb{Z})_\tau$, the expansion of $1/\Phi_{10}$ and $1/\Phi_{k-2}$ for $N = 2, 3, 5, 7$ are manifestly invariant under permutations of $q_1, q_2, q_3$. In contrast, the expansion of $1/\tilde{\Phi}_{k-2}$ is only invariant under permutations of $q_1$ and $q_3$.

## A.5   Fourier-Jacobi coefficients and meromorphic Jacobi forms

Given a meromorphic Siegel modular form $1/\Phi(\rho, \sigma, v)$ of weight $-w$, the Fourier expansion with respect to $\sigma$

$$1/\Phi(\rho, \sigma, v) = \sum_{m \gg -\infty} \psi_m(\rho, v)\, e^{2\pi i m\sigma} \tag{A.51}$$

gives rise to an infinite series of meromorphic Jacobi forms $\psi_m(\rho, v)$ of fixed weight $w$ and increasing index $m$. If $\Phi$ is modular under the full Siegel modular group, then $m \in \mathbb{Z}$ and $\psi_m$ is a Jacobi form for the full Jacobi group $SL(2,\mathbb{Z}) \ltimes \mathbb{Z}^2$, i.e. it satisfies

$$\psi_m(\rho, v + \lambda\rho + \mu) = e^{-2\pi i m(\lambda^2\rho + 2\lambda v)}\psi_m(\rho, v), \tag{A.52}$$

$$\psi_m\left(\frac{a\rho + b}{c\rho + d}, \frac{v}{c\rho + d}\right) = (c\rho + d)^w\, e^{\frac{2\pi i m c v^2}{c\rho + d}}\psi_m(\rho, v) \tag{A.53}$$

for all integers $a, b, c, d, \lambda, \mu$ such that $ad - bc = 1$. If $\Phi$ is modular under a congruence subgroup $\Gamma \subset Sp(4,\mathbb{Z})$, then

1. for $\Gamma = \Gamma_{2,0}(N)$, then $m \in \mathbb{Z}$ and $\psi_m$ is a Jacobi form for the Jacobi group $\Gamma_0(N) \ltimes \mathbb{Z}^2$, i.e. it satisfies (A.52),(A.53) for all integers $a, b, c, d, \lambda, \mu$ such that $ad - bc = 1$ and $c = 0 \bmod N$

2. For $\Gamma = \tilde{\Gamma}_{2,0}(N)$, then $\psi_m$ is a Jacobi form for $\Gamma^0(N) \ltimes \mathbb{Z}^2$, i.e. it satisfies (A.52),(A.53) for all integers $a, b, c, d, \lambda, \mu$ such that $ad - bc = 1$ and $b = 0 \bmod N$;

3. For $\Gamma = \hat{\Gamma}_{2,0}(N)$, then $m \in \mathbb{Z}/N$ and $\psi_m$ is a Jacobi form for $\Gamma_0(N) \ltimes (N\mathbb{Z} \times \mathbb{Z})$ satisfies (A.52),(A.53) for all integers $a, b, c, d, \lambda, \mu$ such that $ad - bc = 1$, $c = 0 \bmod N$ and $\lambda = 0 \bmod N$ (examples of Jacobi forms of index $n/N$ with these periodicity properties are given by $\phi(N\rho, v)$ where $\phi(\rho, v)$ is an ordinary Jacobi form of index $n$ under the full Jacobi group).

In particular, the Fourier-Jacobi expansion of the inverse of the Igusa cusp form is given by [93, (5.16)],

$$\frac{1}{\Phi_{10}} = \frac{1}{\phi_{-2,1}\Delta}q_\sigma^{-1} + 24\frac{\mathcal{P}}{\Delta} + \frac{9\phi_{0,1}^2 + 3E_4\phi_{-2,1}^2}{4\phi_{-2,1}\Delta}q_\sigma + \mathcal{O}(q_\sigma^2)\,, \tag{A.54}$$

where

$$\mathcal{P}(\rho, v) = \frac{\phi_{0,1}}{12\phi_{-2,1}} = \frac{1}{(2\pi i)^2}\left[-\partial_v^2\log\vartheta_1(\rho, v) + 2\pi i\partial_\rho\log\eta^2\right] \tag{A.55}$$

is (up to a factor $(2\pi i)^2$) the Weierstrass function, a weak Jacobi form of weight 2 and index 0.

In the case of CHL orbifolds with $N = 2, 3, 5, 7$, it will be useful to introduce $\hat{\Phi}_{k-2}$, the image of $\Phi_{k-2}$ under an inversion $S_\sigma$,

$$\hat{\Phi}_{k-2}(\Omega) = (i\sqrt{N})^k\sigma^{-(k-2)}\Phi_{k-2}(S_\sigma\circ\Omega) = \tilde{\Phi}_{k-2}(\Omega)\big|_{\rho\leftrightarrow\sigma}\,, \tag{A.56}$$

where we chose the normalization such that $\hat{\Phi}_{k-2} \sim -4\pi^2 v^2\Delta_k(\rho)\Delta_k(\sigma/N)$ near the divisor $v = 0$. The Fourier-Jacobi expansion of $\Phi_{k-2}$ and $\hat{\Phi}_{k-2}$ is given by

$$\frac{1}{\Phi_{k-2}} = \frac{\eta^6(\rho)}{\Delta_k(\rho)\vartheta_1^2(\rho, v)}q_\sigma^{-1} + \psi_0 + \mathcal{O}(q_\sigma)\,, \tag{A.57}$$

$$\frac{1}{\hat{\Phi}_{k-2}} = -\frac{\eta^6(N\rho)}{\Delta_k(\rho)\vartheta_1^2(N\rho, v)}q_\sigma^{-1/N} + \hat{\psi}_0 + \mathcal{O}(q_\sigma^{1/N})\,, \tag{A.58}$$

where $\vartheta_1(\rho, v) = \sum_{n\in\mathbb{Z}}(-1)^n q^{\frac{1}{2}(n-\frac{1}{2})^2}y^{n-\frac{1}{2}}$ (note that it differs from $\vartheta\big[\begin{smallmatrix}1\\1\end{smallmatrix}\big](\rho, v)$ by a factor of i) and

$$\begin{aligned}
\psi_0 &= \frac{k\mathcal{P}(\rho, v)}{\Delta_k(\rho)} + \frac{k}{12(N-1)}\frac{N^2 E_2(N\rho) - N E_2(\rho)}{\Delta_k(\rho)}\,, \\
\hat{\psi}_0 &= \frac{k\mathcal{P}(N\rho, v)}{\Delta_k(\rho)} + \frac{k}{12(N-1)}\frac{E_2(\rho) - N E_2(N\rho)}{\Delta_k(\rho)}\,.
\end{aligned} \tag{A.59}$$

Now, unlike holomorphic or weak Jacobi forms, a meromorphic Jacobi form $\psi_m(\rho, v)$ of index $m > 0$ and weight $w$ in general do not have a theta series decomposition, unless it happens to be holomorphic in the variable $v$. Instead, it was shown in [93, 94] that it can be decomposed into the sum of a polar part and a finite part,

$$\psi_m(\rho, v) = \psi_m^F(\rho, v) + \psi_m^P(\rho, v)\,, \tag{A.60}$$

where the finite part $\psi_F$ is holomorphic in $z$ and has a theta series decomposition,

$$\psi_m^F(\rho, v) = \frac{c_k(m)}{\Delta_k(\rho)}\sum_{\ell\bmod 2m}h_{m,\ell}(\rho)\vartheta_{m,\ell}(\rho, v)\,, \tag{A.61}$$

where

$$\vartheta_{m,\ell}(\rho, v) = \sum_{s\in\mathbb{Z}}q^{(\ell+2ms)^2/4m}y^{\ell+2ms}\,, \tag{A.62}$$

are the standard theta series transforming in the Weil representation of dimension $2m$ while the polar part is a linear combination of Appell–Lerch sums which match the poles of $\psi_m(\rho, v)$ in the $v$ variable. Since Appell–Lerch sums transform inhomogeneously under modular transformations, so does the finite part $\psi_m^F$, which implies that $h_{m,\ell}$ transform as a vector-valued

mock modular form of weight $\frac{3}{2}-k$. In the case at hand, it follows from (A.44) that $\psi_m(\rho, v)$ has a double pole at $v = 0 \bmod \mathbb{Z} + \rho\mathbb{Z}$ with coefficient proportional to $c_k(m)/\Delta_k(\rho)$, where $c_k(m)$ are the Fourier coefficients of $1/\Delta_k(\sigma)$, so

$$\psi_m^P(\rho, v) = \frac{c_k(m)}{\Delta_k(\rho)} \mathcal{A}_m(\rho, v) \,, \tag{A.63}$$

where $\mathcal{A}_m(\rho, v)$ is the standard Appell–Lerch sum [93]

$$\mathcal{A}_m(\rho, v) = \sum_{s \in \mathbb{Z}} \frac{q^{ms^2+s} y^{2ms+1}}{(1-q^s y)^2} \,. \tag{A.64}$$

The latter satisfies the elliptic property (A.52) but not the modular property (A.53). However, it admits a non-holomorphic completion term

$$\mathcal{A}_m^{\star}(\rho, v) = m \sum_{\ell \bmod 2m} \left[ \frac{\overline{\vartheta_{m,\ell}(\rho)}}{2\pi\sqrt{m\rho_2}} - \sum_{\lambda \in \mathbb{Z}+\frac{\ell}{2m}} |\lambda| \operatorname{erfc}\!\left(2|\lambda|\sqrt{\pi m\rho_2}\right) q^{-m\lambda^2} \right] \vartheta_{m,\ell}(\rho, v), \tag{A.65}$$

such that $\widehat{\mathcal{A}}_m \equiv \mathcal{A}_m + \mathcal{A}_m^{\star}$ transforms like a Jacobi form of weight 2 and index $m$, although it is no longer holomorphic in the $\rho$ and $v$ variables. Consequently, both

$$\widehat{\psi}_m^P(\rho, v) = \psi_m^P + \frac{c_k(m)}{\Delta_k(\rho)} \mathcal{A}_m^*(\rho, v) \qquad \text{and} \qquad \widehat{\psi}_m^F(\rho, v) = \psi_m^F - \frac{c_k(m)}{\Delta_k(\rho)} \mathcal{A}_m^*(\rho, v) \tag{A.66}$$

transform like Jacobi forms of weight $2-k$ and index $m$, although neither is holomorphic in the $\rho$ and $v$ variables. Moreover, $\widehat{\psi}_m^F(\rho, v)$ has a theta series decomposition similar to (A.61) with coefficients

$$\widehat{h}_{m,\ell}(\rho) = h_{m,\ell}(\rho) - m \left[ \frac{\overline{\vartheta_{m,\ell}(\rho)}}{2\pi\sqrt{m\rho_2}} - \sum_{\lambda \in \mathbb{Z}+\frac{\ell}{2m}} |\lambda| \operatorname{erfc}\!\left(2|\lambda|\sqrt{\pi m\rho_2}\right) q^{-m\lambda^2} \right], \tag{A.67}$$

transforming as a vector-valued modular form of weight $\frac{3}{2}-k$. By Taylor expanding the denominator, we can rewrite (A.64) as an indefinite theta series of signature $(1,1)$,

$$\mathcal{A}_m(\rho, v) = \frac{1}{2} \sum_{s,\ell \in \mathbb{Z}} \ell \left[\operatorname{sign}(s+u_2) + \operatorname{sign}\ell\right] q^{ms^2+\ell s} y^{2ms+\ell} \,. \tag{A.68}$$

Similarly, its modular completion can be written as an indefinite theta series,

$$\widehat{\mathcal{A}}_m(\rho, v) = \frac{1}{2} \sum_{s,\ell \in \mathbb{Z}} \ell \left[\operatorname{sign}(s+u_2) + \frac{\sqrt{m}}{\pi\ell\sqrt{\tau_2}} F\!\left(\ell\sqrt{\frac{\pi\tau_2}{m}}\right)\right] q^{ms^2+\ell s} y^{2ms+\ell} \,, \tag{A.69}$$

where

$$F(x) = \sqrt{\pi}x \operatorname{erf}(x) + e^{-x^2} \tag{A.70}$$

is a smooth function which asymptotes to $\sqrt{\pi}|x|$ at large $|x|$ [95].

For meromorphic Jacobi forms of index $m = 0$, the decomposition (A.60) still holds, but the finite part $\psi_0^F$ is now independent of $z$, while the non-holomorphic completion term of the Appell–Lerch sum $\mathcal{A}_0(\rho, v)$ reduces to $\mathcal{A}_0^* = 1/(4\pi\rho_2)$. The simplest example, relevant for the present work, is the (rescaled) Weierstrass function (A.55), which decomposes into

$$\mathcal{P}(\rho, v) = \frac{E_2}{12} + \sum_{s \in \mathbb{Z}} \frac{q^s y}{(1-q^s y)^2} = \frac{\hat{E}_2}{12} + \left(\frac{1}{4\pi\rho_2} + \sum_{s \in \mathbb{Z}} \frac{q^s y}{(1-q^s y)^2}\right). \tag{A.71}$$

In particular, it follows from this decomposition and from (A.88) (with $L = 0$) that the integral over the elliptic curve $v \in \mathcal{E}$ is given by

$$\int_{\mathcal{E}} \mathcal{P}(\rho, v) \frac{\mathrm{d}v \mathrm{d}\bar{v}}{2i\rho_2} = \int_{[0,1]^2} \mathrm{d}u_1 \mathrm{d}u_2 \, \mathcal{P}(\rho, u_1 + \rho u_2) = \frac{\hat{E}_2}{12} \,, \tag{A.72}$$

which is non-holomorphic in $\rho$ as a consequence of the pole of $\mathcal{P}(\rho, v)$ at $v = 0$. From this, it follows in particular that the average values of the zero-th Fourier-Jacobi modes (A.59) of $1/\Phi_{k-2}$ and $1/\hat{\Phi}_{k-2}$ with respect to $v$ are given by

$$\begin{aligned}
\int_{[0,1]^2} \mathrm{d}u_1 \, \mathrm{d}u_2 \, \psi_0 &= \frac{k}{12(N-1)} \frac{N^2 \hat{E}_2(N\rho) - \hat{E}_2(\rho)}{\Delta_k(\rho)} \,, \\
\int_{[0,1] \times [0,N]} \mathrm{d}u_1 \, \mathrm{d}u_2 \, \hat{\psi}_0 &= \frac{k}{12(N-1)} \frac{\hat{E}_2(\rho) - \hat{E}_2(N\rho)}{\Delta_k(\rho)} \,.
\end{aligned} \tag{A.73}$$

For negative index $m < 0$, it turns out that any meromorphic Jacobi form $\psi$ can be expressed as a linear combination of iterated derivative of a modified Appell-Lerch sum, (here $y = e^{2\pi i z}, w = e^{2\pi i u}$) [96]

$$F_M(z, u; \tau) = (y/w)^M \sum_{s \in \mathbb{Z}} \frac{w^{-2Ms} q^{Ms(s+1)}}{1 - q^s y/w} \,. \tag{A.74}$$

The latter transforms as a Jacobi form of index $M = -m$ in $u$ and has a simple pole at $u - z \in \mathbb{Z} + \tau\mathbb{Z}$, with residue $1/(2\pi i)$ at $u = z$. If $S$ denotes the set of poles of $\psi(z)$ in a fundamental domain of $\mathbb{C}/(\mathbb{Z} + \tau\mathbb{Z})$, and $D_{n,u}$ are the Laurent coefficients of $\psi$ at $z = u$, then Theorem 1.1 in [96] states that

$$\psi(z) = -\sum_{u \in S} \sum_{n \geq 0} \frac{D_{n,u}}{(2\pi i)^{n-1}(n-1)!} \left[ \partial_v^{n-1} F_{-m}(z, v) \right]_{v=u} \,. \tag{A.75}$$

For the case of interest in this paper, the leading Fourier-Jacobi coefficient $\psi_{-1} = \frac{1}{\eta^{18} \theta_1^2(z)}$ of $1/\Phi_{10}$ has a double pole at $z = 0$ with residue $1/\Delta$, hence

$$\psi_{-1} = \frac{1}{\Delta} \frac{\partial_u}{2\pi i} F_1(z, u; \tau)\big|_{u=0} = -\frac{1}{\Delta} \sum_{s \in \mathbb{Z}} \left[ \frac{y \, q^{s^2+s}}{(1-q^s y)^2} + \frac{2s y \, q^{s^2+s}}{1 - q^s y} \right] \,. \tag{A.76}$$

Note that this plays the role of $\psi_{-1}^P$, while $\psi_{-1}^F$ vanishes. The modified Appell-Lerch sum can be written as an indefinite theta series,

$$\psi_{-1} = -\frac{1}{\Delta} \sum_{s, \ell \in \mathbb{Z}} \left[ (2s + \ell) \frac{\mathrm{sign}\ell + \mathrm{sign}(u_2 + s)}{2} - \frac{1}{4\pi\rho_2} \delta(u_2 + s) \right] q^{s^2 + \ell s} y^\ell \,, \tag{A.77}$$

where $\mathrm{sign}\ell$ is interpreted as $-1$ for $\ell = 0$. To see that this formula is consistent with the quasi-periodicity (A.52), note that under $(y, s, \ell) \to (yq, s-1, \ell+2)$, (A.77) becomes

$$-\frac{1}{\Delta} \sum_{s, \ell \in \mathbb{Z}} \left[ (2s + \ell) \frac{\mathrm{sign}(\ell + 2) + \mathrm{sign}(u_2 + s)}{2} - \frac{1}{4\pi\rho_2} \delta(u_2 + s) \right] q^{s^2 + \ell s + 1} y^{\ell+2} \,. \tag{A.78}$$

This differs from (A.77) (up to the automorphy factor $qy^2$) only due to the terms $\ell = 0$ and $\ell = -1$, but those two terms leads to a vanishing contribution,

$$-\frac{1}{2} \sum_{s \in \mathbb{Z}} \left[ (2s-1) q^{s(s-1)+1} y + 2s \, q^{s^2+1} y^2 \right] = 0 \,. \tag{A.79}$$

Shifting $\ell$ to $\ell - 2s$, (A.77) may be written equivalently as

$$\psi_{-1} = -\frac{1}{\Delta} \sum_{s,\ell \in \mathbb{Z}} \left[ \ell \, \frac{\text{sign}(\ell - 2s) + \text{sign}(u_2 + s)}{2} - \frac{1}{4\pi\rho_2} \delta(u_2 + s) \right] q^{-s^2 + \ell s} \, y^{\ell - 2s} \,, \qquad \text{(A.80)}$$

which resembles the Appell-Lerch sum (A.68) for $m = -1$, except for the replacement of $\text{sign}\ell$ by $\text{sign}(\ell - 2s)$. Of course, the Appell-Lerch sum $\mathcal{A}_{-1}$ would be divergent, while the modified Appell-Lerch sum is absolutely convergent. Similarly, for CHL orbifolds, the leading Fourier-Jacobi coefficient of $1/\Phi_{k-2}$ is given by the same Eq. (A.80) with $\Delta$ replaced by $\Delta_k$.

## A.6 Fourier coefficients and local modular forms

In this section we shall use the decomposition (A.60) to infer the Fourier coefficients of $1/\Phi_{k-2}$ and $1/\tilde{\Phi}_{k-2}$ in the limit $\Omega_2 \to i\infty$. Starting with the maximal rank case, and assuming that $\sigma_2 \gg \rho_2, v_2$, we find

$$\begin{aligned}
C(n, m, L; \Omega_2) &= \int_{[0,1]^3} d^3\Omega_1 \, \frac{e^{-2\pi i(n\rho + Lv + m\sigma)}}{\Phi_{10}(\rho, \sigma, v)} \\
&= C^F(n, m, L) + c(m) \int_{[0,1]^2} d\rho_1 \, dv_1 \, \frac{e^{-2\pi i(n\rho + Lv)}}{\Delta(\rho)} \, \mathcal{A}_m(\rho, v) \,,
\end{aligned} \qquad \text{(A.81)}$$

where $C^F(n, m, L) = \int_{[0,1]^2} d\rho_1 \, dv_1 \, \psi_m^F(\rho, v) \, e^{-2\pi i(n\rho + Lv)}$ are the Fourier coefficients of the finite part of $\psi_m$. To compute the integral in the second line of (A.81), we Fourier expand $1/\Delta(\rho) = \sum_{M \geq -1} c(m) q^m$ and $\mathcal{A}_m(\rho, v)$ using the representation (A.68), and integrate term by term with respect to $v_1$, obtaining

$$\frac{1}{2} \sum_{s,\ell \in \mathbb{Z}} c(m) \, c(n - Ls + ms^2)(L - 2ms) \left[ \text{sign}(u_2 + s) + \text{sign}(L - 2ms) \right], \qquad \text{(A.82)}$$

where we have used $\ell = L - 2ms, M = n - ms^2 - \ell s$. However, while this naive manipulation lead to the correct result for generic $u_2$, it turns out to miss a distributional part localized at $u_2 \in \mathbb{Z}$, originating from the poles of $\mathcal{A}_m(\rho, v)$ at $q^s e^{2\pi i v} = 1$.

To compute this distribution, let us first consider the contribution from the term $s = 0$ in the sum (A.64). Upon expanding

$$\frac{y}{(1-y)^2} = \begin{cases} \sum_{k \geq 1} k y^k \,, & |y| < 1 \\ \sum_{k \geq 1} k y^{-k} \,, & |y| > 1 \end{cases}, \qquad \text{(A.83)}$$

one would be tempted to conclude that the integral $\int_0^1 dv_1 \, \frac{y}{(1-y)^2}$ vanishes. However, we claim that instead,

$$\int_0^1 dv_1 \, \frac{y}{(1-y)^2} = -\frac{1}{4\pi} \delta(v_2) \,. \qquad \text{(A.84)}$$

To see this, we first consider first the single pole function $\frac{1}{2} \frac{y+1}{y-1}$, with Fourier expansion

$$\frac{1}{2} \frac{y+1}{y-1} = -\sum_{\ell \in \mathbb{Z}} \frac{\text{sign}(\ell) + \text{sign}(v_2)}{2} y^\ell \,, \qquad \text{(A.85)}$$

with the understanding that $\text{sign}(0) = 0$. We claim that this identity is valid at the distributional level. As a check, using the Euler formula representation for (A.85) and acting with an anti-holomorphic derivative on each term (recalling that $\partial_{\bar{v}} \frac{1}{v} = \pi \delta(v_1) \delta(v_2)$), we get

$$-\frac{1}{2\pi i} \frac{\partial}{\partial \bar{v}} \left( \frac{1}{2} \frac{y+1}{y-1} \right) = -\frac{1}{2\pi i} \frac{\partial}{\partial \bar{v}} \left( \sum_{\ell \in \mathbb{Z}} \frac{1}{2\pi i(v - \ell)} \right) = \frac{1}{4\pi} \delta(v_2) \sum_{\ell \in \mathbb{Z}} \delta(v_1 - \ell) = \frac{1}{4\pi} \delta(v_2) \sum_{\ell \in \mathbb{Z}} y^\ell \,. \qquad \text{(A.86)}$$

The right-hand side is also what one gets by acting with $\partial_{\bar{v}} = \frac{1}{2}(\partial_{v_1} + i\partial_{v_2})$ on each term in the Fourier series (A.85), noting that $\text{sign}'(v_2) = 2\delta(v_2)$.

The double pole distribution (A.83) is obtained by acting with a holomorphic derivative on (A.85), therefore admits the Fourier expansion

$$\frac{y}{(1-y)^2} = -\frac{1}{2\pi i}\frac{\partial}{\partial v}\Big(\frac{1}{2}\frac{y+1}{y-1}\Big) = \sum_{\ell\in\mathbb{Z}}\frac{|\ell|+\text{sign}(v_2)\ell}{2}y^\ell - \frac{1}{4\pi}\delta(v_2)\sum_{\ell\in\mathbb{Z}}y^\ell . \tag{A.87}$$

In particular, integrating over $v_1$ we reach (A.84).[30] More generally, the same argument shows that for any $s$,

$$\frac{q^s y}{(1-q^s y)^2} = \sum_{\ell\in\mathbb{Z}}\Big(\frac{|\ell|+\ell\,\text{sign}(v_2+s\rho_2)}{2} - \frac{1}{4\pi}\delta(v_2+s\rho_2)\Big)q^{\ell s}y^\ell . \tag{A.88}$$

Using this identity, we find that the naive result (A.82) misses an additional term supported at $u_2 = v_2/\rho_2 \in \mathbb{Z}$,

$$-\frac{1}{4\pi}\sum_{s,\ell\in\mathbb{Z}}c(m)c(n-Ls+ms^2)\delta(v_2+s\rho_2) . \tag{A.89}$$

However, this still cannot be the full Fourier coefficient $C(n,m,L;\Omega_2)$, since the latter must be invariant under the action (A.14) of $GL(2,\mathbb{Z})$. Instead, both (A.82) and (A.89) are invariant under the subgroup $\Gamma_\infty$ which preserves the cusp $\sigma_2 = \infty$, where $\big(\begin{smallmatrix}1 & s\\ 0 & 1\end{smallmatrix}\big)$ acts by sending $(n,m,L) \to (n-Ls+ms^2, mL-2ms)$. To restore invariance under the full $GL(2,\mathbb{Z})$ group, we may therefore replace the sum over $s \in \mathbb{Z}$ by a sum over all $\gamma \in GL(2,\mathbb{Z})/\text{Di}_4$, obtaining

$$\begin{aligned}C(n,m,L;\Omega_2) = {}& C^F(n,m,L) \\ & + \sum_{\gamma\in GL(2,\mathbb{Z})/\text{Dih}_4}\Big[c(m)c(n)\Big(\frac{1}{2}L(\text{sgn}L+\text{sgn}v_2) - \frac{1}{4\pi}\delta(v_2)\Big)\Big]\big|_\gamma + \dots .\end{aligned} \tag{A.90}$$

Here, $\text{Dih}_4$ denotes the dihedral group generated by the matrices $\big(\begin{smallmatrix}1 & 0\\ 0 & -1\end{smallmatrix}\big)$ and $\big(\begin{smallmatrix}0 & 1\\ 1 & 0\end{smallmatrix}\big)$, which stabilizes the locus $v_2 = 0$, and the dots denotes possible additional contributions which are not visible in the limit $|\Omega_2| \to \infty$. The action of $\gamma = \big(\begin{smallmatrix}p & q\\ r & s\end{smallmatrix}\big) \in GL(2,\mathbb{Z})$ on the quantities $m,n,L,v_2$ appearing in the bracket is given by

$$\begin{aligned}\hat{n} &\mapsto s^2 n + q^2 m - qsL, \qquad \hat{m} \mapsto r^2 n + p^2 m - prL , \tag{A.91}\\ \hat{L} &\mapsto -2rs\,n - 2pq\,m + \frac{ps+qr}{2}L , \tag{A.92}\\ \hat{v}_2 &\mapsto \text{tr}\big(\big(\begin{smallmatrix}0 & 1/2\\ 1/2 & 0\end{smallmatrix}\big)\gamma^\intercal\Omega_2\gamma\big) = pq\,\rho_2 + rs\,\sigma_2 + (ps+qr)v_2 . \tag{A.93}\end{aligned}$$

Using the same reasoning, we find the Fourier coefficients of $1/\Phi_{k-2}$, which must be invariant under $GL(2,\mathbb{Z})$,

$$\begin{aligned}C_{k-2}(n,m,L;\Omega_2) = {}& C^F_{k-2}(n,m,L) \\ & + \sum_{\gamma\in GL(2,\mathbb{Z})/\text{Dih}_4}\Big[c_k(\hat{m})c_k(\hat{n})\Big(\frac{1}{2}\hat{L}\big(\text{sign}\hat{L}+\text{sign}\hat{v}_2\big) - \frac{1}{4\pi}\delta(\hat{v}_2)\Big)\Big]_\gamma + \dots . \end{aligned} \tag{A.94}$$

---

[30]It is worth cautioning the reader that regularizing the double pole by point splitting would instead produce the same delta distribution with coefficient $-1/(2\pi)$. This however would be inconsistent with modular invariance, e.g. when computing the average of the Weierstrass function in (A.72).

For the Fourier coefficients of $1/\tilde{\Phi}_{k-2}$, which must be invariant under $\Gamma_0(N)$, we find instead

$$\begin{aligned}
\widetilde{C}_{k-2}(n,m,L;\Omega_2) = {} & \widetilde{C}_{k-2}^F(n,m,L) \\
& + \sum_{\gamma \in \Gamma_0(N)/\mathbb{Z}_2} \left[ c_k(N\hat{m})\, c_k(\hat{n}) \left[ \frac{1}{2}\hat{L}\left( \text{sign}\hat{L} + \text{sign}\hat{v}_2 \right) - \frac{1}{4\pi}\delta(\hat{v}_2) \right] \right]_\gamma + \dots \,.
\end{aligned} \quad (A.95)$$

It is important to note that the identities (A.90),(A.94),(A.95) are only valid when $|\Omega_2|$ is large enough such that the integration contour $[0,1]^3 + i\Omega_2$ does not cross any pole for generic values of $\Omega_2$, and only crosses quadratic divisors (A.6) with $n_2 = 0$ on real-codimension one loci. When $|\Omega_2| < 1/(4n_2^2)$ with $|n_2| \geq 1$, the contour crosses the the quadratic divisor (A.6) for generic values of $\Omega_2$, and the integral on the first line of (A.81) is no longer well-defined. We leave it as an interesting open problem to define the Fourier coefficient $C(n,m,L;\Omega_2)$ of $1/\Phi_{10}$ (or its analogue for $1/\Phi_{k+2}$ and $1/\tilde{\Phi}_{k-2}$) in the region where $|\Omega_2| \leq 1/4$.

# B  Perturbative contributions to 1/4-BPS couplings

In this section, we compute the one-loop and two-loop contributions to the coefficient of the $\nabla^2 F^4$ coupling in the low-energy effective action in heterotic CHL orbifolds. In both cases we start with the maximal rank case, *i.e.* heterotic string compactified on a torus $T^d$, and then turn to the simplest heterotic CHL orbifolds with $N = 2,3,5,7$.

## B.1  One-loop $\nabla^2 F^4$ and $\mathcal{R}^2 F^2$ couplings

### B.1.1  Maximal rank case

In heterotic string compactified on a torus $T^d$, the one-loop contribution to the coefficient of the $\nabla^2 F^4$ coupling in the low-energy effective action can be extracted from the four-gauge boson one-loop amplitude, given up to an overal tensorial factor by [42]

$$\begin{aligned}
\mathcal{A}_{abcd}^{(1)} = {} & \frac{1}{(2\pi i)^4} \int_{\mathcal{F}_1} \frac{d\rho_1 d\rho_2}{\rho_2^2} \frac{1}{\Delta} \int_{\mathcal{E}^4} \prod_{i=1}^4 \frac{dz_i d\bar{z}_i}{2i\rho_2} (\chi_{12}\chi_{34})^{\alpha's} (\chi_{13}\chi_{24})^{\alpha't} (\chi_{14}\chi_{23})^{\alpha'u} \\
& \times \langle J_a(z_1) J_b(z_2) J_c(z_3) J_d(z_4) \rangle \,,
\end{aligned} \quad (B.1)$$

where $\chi_{ij} = e^{g(\rho,z_i-z_j)}$ and $g(\rho,z) = -\log|\theta_1(\rho,z)/\eta|^2 + \frac{2\pi}{\rho_2}(\text{Im}z)^2$ is the scalar Green function on the elliptic curve $\mathcal{E}$ with modulus $\rho$. The four-point function of the currents evaluates to

$$\begin{aligned}
\langle J_a(z_1) J_b(z_2) J_c(z_3) J_d(z_4) \rangle = {} & \Gamma_{\Lambda_{d+16,d}}[P_{abcd}] - \frac{1}{4\pi^2}\left( \delta_{ab}\Gamma_{\Lambda_{d+16,d}}[P_{cd}]\partial^2 g(z_1-z_2) + 5\,\text{perms} \right) \\
& + \frac{1}{16\pi^4}\left( \delta_{ab}\delta_{cd}\,\Gamma_{\Lambda_{d+16,d}}[1]\partial^2 g(z_1-z_2)\partial^2 g(z_3-z_4) + 2\,\text{perms} \right) \,,
\end{aligned} \quad (B.2)$$

where $P_{ab}$ and $P_{abcd}$ are quadratic and quartic polynomials, respectively, in the projected lattice vector $Q_{La} = p_{La}{}^{\mathcal{I}}Q_{\mathcal{I}} \in \Gamma_{d+16,d}$ arising from the zero-mode of the currents,

$$\begin{aligned}
P_{ab} = {} & Q_{La}Q_{Lb} - \frac{\delta^{ab}}{4\pi\rho_2} \,, \\
P_{abcd} = {} & Q_{La}Q_{Lb}Q_{Lc}Q_{Ld} - \frac{3}{2\pi\rho_2}\delta_{(ab}Q_{Lc}Q_{Ld)} + \frac{3}{16\pi^2\rho_2^2}\delta_{(ab}\delta_{cd)} \,,
\end{aligned} \quad (B.3)$$

and for any polynomial $P$ in $Q_{La}$ and integer lattice $\Lambda_{p,q}$ of signature $(p,q)$, we denote

$$\Gamma_{\Lambda_{p,q}}[P] = \rho_2^{q/2} \sum_{Q \in \Lambda_{p,q}} P(Q_{La}) e^{i\pi[\rho Q_L^2 - \bar{\rho} Q_R^2]} . \tag{B.4}$$

Upon expanding in powers of $\alpha'$, the leading term reproduces the one-loop contribution to the $F^4$ coupling,

$$F_{abcd}^{(1)} = \text{R.N.} \int_{\mathcal{F}_1} \frac{d\rho_1 d\rho_2}{\rho_2^2} \frac{\Gamma_{\Lambda_{d+16,d}}[P_{abcd}]}{\Delta(\rho)} , \tag{B.5}$$

where R.N. denotes the regularization procedure introduced in [97–99], which is needed to make sense of the divergent integral when $d \geq 6$ (we return to this point at the end of this subsection). Equivalently, (B.5) may be written as [47]

$$F_{abcd}^{(1)} = \text{R.N.} \int_{\mathcal{F}_1} \frac{d\rho_1 d\rho_2}{\rho_2^2} \frac{\partial^4}{(2\pi i)^4 \partial y^a \partial y^b \partial y^c \partial y^d} \frac{\Gamma_{\Lambda_{d+16,d}}(y)}{\Delta(\rho)}\bigg|_{y=0} , \tag{B.6}$$

where $\Gamma_{\Lambda_{p,q}}(y)$ is the partition function of the compact bosons deformed by the current $y_a J^a$ integrated along the $A$-cycle of the elliptic curve,

$$\Gamma_{\Lambda_{p,q}}(y) = \rho_2^{q/2} \sum_{Q \in \Lambda_{p,q}} e^{i\pi[\rho Q_L^2 - \bar{\rho} Q_R^2] + 2\pi i Q_L \cdot y + \frac{\pi(y \cdot y)}{2\rho_2}} . \tag{B.7}$$

At next to leading order in $\alpha'$, the term linear in the Mandelstam variables $s, t, u$ reduces to

$$G_{ab,cd}^{(1)} = \int_{\mathcal{F}_1} \frac{d\rho_1 d\rho_2}{\rho_2^2} \frac{1}{\Delta} \int_{\mathcal{E}^4} \prod_{i=1}^4 \frac{dz_i d\bar{z}_i}{2i\rho_2} \left[ g(z_1 - z_2) \partial^2 g(z_1 - z_2) \delta_{ab} \Gamma_{\Lambda_{d+16,d}}[P_{cd}] + 5\,\text{perms} \right] , \tag{B.8}$$

since all other terms at this order are total derivatives with respect to $z_i$. The integral over $z$ can be computed by using the Poincaré series representation of the Green function,

$$g(\rho, z) = \frac{1}{\pi} \sideset{}{'}\sum_{(m,n) \in \mathbb{Z}^2} \frac{\rho_2}{|m\rho + n|^2} e^{\frac{\pi}{\rho_2}[\bar{z}(m\rho + n) - z(m\bar{\rho} + n)]} , \tag{B.9}$$

leading to

$$\int_{\mathcal{E}} \frac{dz d\bar{z}}{2i\rho_2} g(z - w) \partial^2 g(z - w) = \lim_{s \to 0} \sideset{}{'}\sum_{(m,n) \in \mathbb{Z}^2} \frac{1}{(m\rho + n)^2 |m\rho + n|^{2s}} = \frac{\pi^2}{6} \hat{E}_2 , \tag{B.10}$$

where the sum over $(m, n)$ was regularized à la Kronecker. Up to an overall numerical factor, we therefore find that the one-loop contribution to the coefficient of $\nabla^2 F^4$ coupling for the maximal rank model is given by

$$G_{ab,cd}^{(1)} = \delta_{\langle ab} G_{cd \rangle}^{(d+16,d)} , \qquad G_{ab}^{(p,q)} = \text{R.N.} \int_{\mathcal{F}_1} \frac{d\rho_1 d\rho_2}{\rho_2^2} \frac{\hat{E}_2}{\Delta(\rho)} \Gamma_{\Lambda_{p,q}}[P_{ab}] . \tag{B.11}$$

For $d = 0$, corresponding to either of the $E_8 \times E_8$ or $Spin(32)/\mathbb{Z}_2$ heterotic strings in 10 dimensions, one has

$$\Gamma_{\Lambda_{E_8 \oplus E_8}}[P_{ab}] = \Gamma_{\Lambda_{D_{16}}}[P_{ab}] = \frac{E_4}{12} \left( \hat{E}_2 E_4 - E_6 \right) \delta_{ab} , \tag{B.12}$$

so $G_{ab,cd}^{(1)}$ becomes proportional to the $\text{Tr}F^2 \text{Tr}\mathcal{R}^2$ coupling computed from the elliptic genus [100, C.5], [101], as required by supersymmetry.

### B.1.2 CHL orbifolds

The four-gauge boson amplitude in CHL models with $N = 2, 3, 5, 7$ was obtained in [45, 46]. It was shown in [22, §A] that the one-loop $F^4$ coupling in these models is given by the simple generalization of (B.5), namely

$$F^{(1)}_{abcd} = F^{(d+r-12,d)}_{abcd}, \qquad F^{(p,q)}_{abcd} = \text{R.N.} \int_{\Gamma_0(N)\backslash\mathcal{H}_1} \frac{d\rho_1 d\rho_2}{\rho_2^2} \frac{\Gamma_{\Lambda_{p,q}}[P_{abcd}]}{\Delta_k}, \qquad \text{(B.13)}$$

where $\Delta_k = [\eta(\rho)\eta(N\rho)]^k$ arises from the partition function in the twisted sectors. The same derivation goes through for the $\nabla^2 F^4$ and $\mathcal{R}^2 F^2$ couplings and yields

$$G^{(1)}_{ab,cd} = \delta_{\langle ab} G^{(d+r-12,d)}_{cd\rangle}, \qquad G^{(p,q)}_{ab} = \text{R.N.} \int_{\Gamma_0(N)\backslash\mathcal{H}_1} \frac{d\rho_1 d\rho_2}{\rho_2^2} \frac{\hat{E}_2}{\Delta_k} \Gamma_{\Lambda_{p,q}}[P_{ab}]. \qquad \text{(B.14)}$$

### B.1.3 Regularization of the genus-one modular integrals

As indicated above, the modular integrals (B.13) and (B.14) are divergent when $d \geq 6$ and $d \geq 4$, respectively. We follow the same regularization procedure as in [22, 102] and define them by truncating the integration domain to $\mathcal{F}_{N,\Lambda} = \cup_{\gamma\in\Gamma_0(N)\backslash SL(2,\mathbb{Z})}\gamma \cdot \mathcal{F}_{1,\Lambda}$, where $\mathcal{F}_{1,\Lambda} = \{-\frac{1}{2} < \rho_1 < \frac{1}{2}, |\rho| > 1, \rho_2 < \Lambda\}$ is the truncated fundamental domain for $SL(2,\mathbb{Z})$, and minimally subtracting the divergent terms before taking the limit $\Lambda \to \infty$. Using the fact that the constant terms of $1/\Delta_k$ and $\hat{E}_2/\Delta_k$ are equal to $k$ and $k(1 - \frac{3}{\pi\rho_2}) - 24$, the constant terms of their Fricke dual are $k$ and $k(N - \frac{3}{\pi\rho_2})$ and the constant terms of the Fricke dual of the partition function include an extra factor of $\upsilon N^{\frac{q-8}{2}}$, we get

$$F^{(p,q)}_{abcd} = \lim_{\Lambda\to\infty}\left[\int_{\mathcal{F}_{N,\Lambda}} \frac{d\rho_1 d\rho_2}{\rho_2^2} \frac{\Gamma_{\Lambda_{p,q}}[P_{abcd}]}{\Delta_k} - \frac{3k(1 + \upsilon N^{\frac{q-8}{2}})}{16\pi^2} \frac{\Lambda^{\frac{q-6}{2}}}{\frac{q-6}{2}} \delta_{(ab}\delta_{cd)}\right], \qquad \text{(B.15)}$$

$$G^{(p,q)}_{ab} = \lim_{\Lambda\to\infty}\left[\int_{\mathcal{F}_{N,\Lambda}} \frac{d\rho_1 d\rho_2}{\rho_2^2} \frac{\hat{E}_2}{\Delta_k}\Gamma_{\Lambda_{p,q}}[P_{ab}] - \frac{3k(1 + \upsilon N^{\frac{q-8}{2}})}{4\pi^2} \frac{\Lambda^{\frac{q-6}{2}}}{\frac{q-6}{2}} \delta_{ab} + \frac{k(1 + \upsilon N^{\frac{q-6}{2}}) - 24}{4\pi} \frac{\Lambda^{\frac{q-4}{2}}}{\frac{q-4}{2}} \delta_{ab}\right], \qquad \text{(B.16)}$$

where the terms $\Lambda^{\frac{q-6}{2}}/\frac{q-6}{2}$ and $\Lambda^{\frac{q-4}{2}}/\frac{q-4}{2}$ should be replaced by $\log\Lambda$ when $q = 6$ or $q = 4$, respectively. Note that the second term in (B.16) cancels in the case of the full rank model where $k = 24$. It will be also useful to consider the Fricke dual function to $G^{(p,q)}_{ab}$ for the $N = 2, 3, 5, 7$ models, introduced in (4.57) and whose regularization is given by

$$\varsigma G^{(p,q)}_{ab} = \lim_{\Lambda\to\infty}\left[\int_{\mathcal{F}_{N,\Lambda}} \frac{d\rho_1 d\rho_2}{\rho_2^2} \frac{N\hat{E}_2(N\rho)}{\Delta_k(\rho)}\Gamma_{\Lambda_{p,q}}[P_{ab}] - \frac{3k(1 + \upsilon N^{\frac{q-8}{2}})}{4\pi^2} \frac{\Lambda^{\frac{q-6}{2}}}{\frac{q-6}{2}} \delta_{ab}\right.$$
$$\left. + \frac{k(N + \upsilon N^{\frac{q-8}{2}}) - 24}{4\pi} \frac{\Lambda^{\frac{q-4}{2}}}{\frac{q-4}{2}} \delta_{ab}\right]. \qquad \text{(B.17)}$$

### B.1.4 Differential identities satisfied by genus-one modular integrals

Like the genus-two modular integral $G^{(p,q)}_{ab,cd}$ discussed in §3.3, the genus-one modular integrals (B.15), (B.16) and (B.17) satisfy differential identities with constant source terms in $q = 6$, $q = 4$ determined by regularization techniques using the same paramatrization as for section

**B.1.3.** The equation for the modular integral $F_{abcd}^{(p,q)}$ was calculated in [22, (3.57)], which we reproduce below:

$$
\begin{aligned}
\mathcal{D}_{(e}{}^{\hat{g}}\mathcal{D}_{f)\hat{g}}F_{abcd} =& \tfrac{2-q}{4}\delta_{ef}\,F_{abcd} + (4-q)\delta_{(e|(a}F_{bcd)|f)} \\
& + 3\delta_{(ab}F_{cd)ef} + \frac{15k(1+\frac{v}{N})}{2(4\pi)^2}\delta_{(ab}\delta_{cd}\delta_{ef)}\,\delta_{q,6}\,.
\end{aligned}
$$

Here the volume factor $v$ is either equal to $N$ for the perturbative Narain lattice, or to 1 for the non-perturbative Narain lattice.

The equation satisfied by the genus-one integral $G_{ab}^{(p,q)}$ can be computed using the same techniques described in [22, §3.2] and reads

$$
\begin{aligned}
\mathcal{D}_{(e}{}^{\hat{g}}\mathcal{D}_{f)\hat{g}}G_{ab}^{(p,q)} =& \frac{2-q}{4}\,\delta_{ef}\,G_{ab}^{(p,q)} + \frac{4-q}{2}\,\delta_{e)(a}\,G_{b)(f}^{(p,q)} + \frac{1}{2}\,\delta_{ab}\,G_{ef}^{(p,q)} + 6\,F_{efab}^{(p,q)} \\
& - \frac{3\big((1+\frac{v}{N})k - 24\big)}{8\pi}\delta_{(ef}\delta_{ab)}\,\delta_{q,4} + \frac{9(1+\frac{v}{N})k}{8\pi^2}\delta_{(ef}\delta_{ab)}\,\delta_{q,6}\,,
\end{aligned}
\tag{B.18}
$$

where the term proportional to $F_{efab}^{(p,q)}$ corresponds to the contribution of the non-holomorphic completion in $\hat{E}_2$, and the two constant contributions of the second line correspond to the boundary contribution after integration by part (see [22, (3.54)]). One checks that the divergent contributions cancel each others, so the equation is valid for the renormalized couplings. For the perturbative lattice with $v = N$, these linear corrections are associated to the mixing between the analytic and the non-analytic components of the amplitude, and are indeed proportional to the corresponding 1-loop divergence coefficient in supergravity [83].

The same analysis for ${}^{\varsigma}G_{ab}^{(p,q)}$ gives

$$
\begin{aligned}
\mathcal{D}_{(e}{}^{\hat{g}}\mathcal{D}_{f)\hat{g}}{}^{\varsigma}G_{ab}^{(p,q)} =& \frac{2-q}{4}\,\delta_{ef}\,{}^{\varsigma}G_{ab}^{(p,q)} + \frac{4-q}{2}\,\delta_{e)(a}\,{}^{\varsigma}G_{b)(f}^{(p,q)} + \frac{1}{2}\,\delta_{ab}\,{}^{\varsigma}G_{ef}^{(p,q)} + 6\,F_{efab}^{(p,q)} \\
& - \frac{3\big((N+v)k - 24\big)}{8\pi}\delta_{(ef}\delta_{ab)}\,\delta_{q,4} + \frac{9(1+\frac{v}{N})k}{8\pi^2}\delta_{(ef}\delta_{ab)}\,\delta_{q,6}\,.
\end{aligned}
\tag{B.19}
$$

## B.2 Two-loop $\nabla^2 F^4$ couplings

### B.2.1 Maximal rank case

At two-loop, the scattering amplitude of four gauge bosons in ten-dimensional heterotic string theory was computed in [43, 44]. Upon compactifying on a torus $T^d$, one obtains

$$
\begin{aligned}
\mathcal{A}_{abcd}^{(2)} =& \int_{\mathcal{F}_2} \frac{d^3\Omega_1\,d^3\Omega_2}{|\Omega_2|^3}\,\frac{1}{\Phi_{10}} \\
& \times \int_{\Sigma^4} \overline{\mathcal{Y}}_S \prod_{i=1}^{4} dz_i\,(\chi_{12}\chi_{34})^{\alpha's}\,(\chi_{13}\chi_{24})^{\alpha't}\,(\chi_{14}\chi_{23})^{\alpha'u}\,\langle J_a(z_1)J_b(z_2)J_c(z_3)J_d(z_4)\rangle\,,
\end{aligned}
\tag{B.20}
$$

where $\Sigma$ is a genus-two Riemann surface with period matrix $\Omega$, $\overline{\mathcal{Y}}_S$ is a specific $(1,1)$ form in each of the coordinates $z_i$ on $\Sigma$ [43, (11.32)],

$$
\mathcal{Y}_S = t\,\Delta(1,2)\,\Delta(3,4) - s\,\Delta(1,4)\,\Delta(2,3)\,,
\tag{B.21}
$$

where $\Delta(z,w) = \omega_1(z)\omega_2(w) - \omega_1(w)\omega_2(z)$, $\chi_{ij} = e^{G(\Omega,z_i-z_j)}$ and $G(\Omega,z)$ is the scalar Green function on $\Sigma$. At leading order in $\alpha'$, $\chi_{ij}$ can be set to one, and similarly to (B.6), the integrated

current correlator $\int_\Sigma J^a(z)\mathrm{d}z\,\overline{\omega_I z}$ can be expressed as a multiple derivative [47]

$$\langle \int_{\Sigma^4} J^a(z_1)J^b(z_2)J^c(z_3)J^d(z_4)\prod_{i=1}^4 \mathrm{d}z_i\,\overline{\omega_I(z_i)}\rangle = \frac{\frac{1}{3}(\varepsilon_{rr'}\varepsilon_{ss'}+\varepsilon_{rs'}\varepsilon_{sr'})\partial^4}{(2\pi\mathrm{i})^4 \partial\, y_a^r\, \partial\, y_b^s\, \partial\, y_c^{r'}\, \partial\, y_d^{s'}}\Gamma^{(2)}_{\Lambda_{d+16,d}}(y)|_{y=0}\,,$$

(B.22)

where $\Gamma^{(2)}_{\Lambda_{d+16,d}}(y)$ is the partition function of the compact bosons deformed by the currents $y_a^r J^a$ integrated along the $r$-th A-cycle of $\Sigma$,

$$\Gamma^{(2)}_{\Lambda_{p,q}}(y) = |\Omega|_2^{q/2} \sum_{Q\in\Lambda_{p,q}^{\otimes 2}} e^{\mathrm{i}\pi Q_{La}^r\,\Omega_{rs}\,Q_L^{s\,a}-\mathrm{i}\pi Q_{R\hat{a}}^r\,\bar{\Omega}_{rs}\,Q_R^{s\,\hat{a}}+2\pi\mathrm{i}Q_{La}^r y_r^a+\frac{\pi}{2}y_r^a\Omega_2^{rs}y^{as}}\,.$$

(B.23)

Evaluating the derivatives explicitly, we obtain the result announced in (2.30) for the two-loop $\nabla^2 F^4$ coupling in the maximal rank case,

$$G_{ab,cd}^{(d,d+16)} = \text{R.N.} \int_{\mathcal{F}_2} \frac{\mathrm{d}^3\Omega_1\,\mathrm{d}^3\Omega_2}{|\Omega_2|^3}\frac{\Gamma^{(2)}_{\Lambda_{d,d+16}}[P_{ab,cd}]}{\Phi_{10}}\,,$$

(B.24)

where $P_{ab,cd}$ is the quartic polynomial defined in (2.31). The regularization procedure needed to make sense of this modular integral when $d \geq 5$ will be discussed in §B.2.4.

In the special case ($d=0$) of the $E_8 \times E_8$ heterotic string in 10 dimensions, and for a suitable choice of indices $ab,cd$, the partition function $\Gamma^{(2)}_{\Lambda_{E_8\times E_8}}[P_{ab;cd}]$ reduces (up to normalization) to $(E_4^{(2)})^2\Psi_2$ where $E_4^{(2)}$ is the holomorphic Eisenstein series of weight 4, which coincides with the Siegel theta series for the lattice $E_8$, and $\Psi_2$ is a non-holomorphic modular form of weight $(2,0)$ given by

$$\Psi_2 = \partial_\rho\Phi\partial_\sigma\Phi - \tfrac{1}{4}(\partial_\nu\Phi)^2\,,\quad \Phi = \log\left[|\Omega_2|^4 E_4^{(2)}\right]\,,$$

(B.25)

in agreement with [44, (5.7)]. This can be viewed as the genus-two counterpart of the genus-one formula (B.12). We shall now discuss the extension of (B.24) to CHL orbifolds, starting with the simplest case $N=2$.

## B.2.2  $\mathbb{Z}_2$ orbifold

The simplest CHL model is obtained by orbifolding the $E_8 \times E_8$ heterotic string on $T^d$ by an involution $\sigma$ exchanging the two $E_8$ factors, and translating by half a period along one circle in $T^d$ [25]. This model was studied in more detail in [103, 104] and revisited in [22, §A.1]. Some aspects of the genus-two heterotic amplitude in this model were discussed in [11] in the context of 1/4-BPS dyon counting, which we shall build on.

Following standard rules, the two-loop amplitude is now a sum over all possible twisted or untwisted periodicity conditions $[h_1 h_2]$ and $[g_1 g_2]$ along the $A$ and $B$ cycles of the genus-two curve $\Sigma$, respectively,

$$\mathcal{A}^{(2)} = \frac{1}{4}\sum_{\substack{h_1,h_2\in\{0,1\}\\g_1,g_2\in\{0,1\}}} \mathcal{A}^{(2)}\begin{bmatrix} h_1 h_2\\ g_1 g_2\end{bmatrix}\,.$$

(B.26)

The untwisted amplitude $\mathcal{A}^{(2)}\begin{bmatrix} 00\\ 00\end{bmatrix}$ coincides with (B.20), restricted on the locus $G_{d+8,d}\subset G_{d+16,d}$ which is invariant under the involution $\sigma$. As in the genus-one case [22, §A.1], it is convenient to further restrict to the locus $G_{d,d}\subset G_{d+8,d}$ where the lattice factorizes as $\Lambda_{d+16,d}=E_8\oplus E_8\oplus I\!I_{d,d}$, and retain from $\mathcal{A}^{(2)}\begin{bmatrix} h_1 h_2\\ g_1 g_2\end{bmatrix}$ the chiral measure for the ten-dimensional string, which we denote by

$$Z_{16}^{(2)}\begin{bmatrix} 00\\ 00\end{bmatrix} = \frac{\left[\Theta_{E_8}^{(2)}(\Omega)\right]^2}{\Phi_{10}}\,.$$

(B.27)

Now, decomposing $p_1^\alpha + p_2^\alpha = 2\Sigma^\alpha + \mathcal{P}^\alpha, p_1^\alpha - p_2^\alpha = 2\Delta^\alpha - \mathcal{P}^\alpha$ for $p_1^\alpha, p_2^\alpha \in \Lambda_{E_8}, \alpha = 1, 2$, the genus-two partition function of the lattice $\Lambda_{E_8 \times E_8}$ appearing in the numerator can be decomposed as

$$\left[\Theta_{E_8}^{(2)}(\Omega)\right]^2 = \sum_{(\mathcal{P}_1, \mathcal{P}_2) \in (\Lambda_{E_8}/2\Lambda_{E_8})^{\otimes 2}} \Theta_{E_8[2],(\mathcal{P}_1,\mathcal{P}_2)}^{(2)}(\Omega)\, \Theta_{E_8[2],(\mathcal{P}_1,\mathcal{P}_2)}^{(2)}(\Omega) \,, \qquad (B.28)$$

where $\Theta_{E_8[2],(\mathcal{P}_1,\mathcal{P}_2)}^{(2)}$ is the genus-two theta series for $\Lambda_{E_8}[2]$:

$$\Theta_{E_8[2],(\mathcal{P}_1,\mathcal{P}_2)}^{(2)}(\Omega) = \sum_{(\Delta^1,\Delta^2) \in \Lambda_{E_8}^{\otimes 2}} e^{2\pi i(\Delta^r - \frac{1}{2}\mathcal{P}^r)\Omega_{rs}(\Delta^s - \frac{1}{2}\mathcal{P}^s)} \,. \qquad (B.29)$$

For $\mathcal{P}_1 = \mathcal{P}_2 = 0, \Theta_{E_8[2],(0,0)}^{(2)}(\Omega) = \Theta_{E_8}^{(2)}(2\Omega)$.

As for the twisted sectors $\begin{bmatrix} h \\ g \end{bmatrix} \equiv \begin{bmatrix} h_1 h_2 \\ g_1 g_2 \end{bmatrix} \neq \begin{bmatrix} 00 \\ 00 \end{bmatrix}$, we use the fact that the $\mathbb{Z}_2$ orbifold blocks of $d$ compact scalars on a Riemann surface of genus 2 are given by [105, 106]

$$\left| \frac{\vartheta^{(2)}[\delta_i^+](0,\Omega)\, \vartheta^{(2)}[\delta_i^-](0,\Omega)}{Z_0(\Omega)^2\, \vartheta_i(0,\tau_{h,g})^2} \right|^d \sum_{Q \in \Lambda_{d,d}} e^{i\pi p_L^2(Q)\tau_{h,g} - i\pi p_R^2(Q)\bar\tau_{h,g}} \,, \qquad (B.30)$$

where $Z_0(\Omega)$ is the inverse of the chiral partition of a (uncompactified, untwisted, unprojected) scalar field on $\Sigma$, and $\tau_{h,g}$ is the Prym period, namely the period of the unique even holomorphic form on the double cover of $\Sigma$, a Riemann surface $\hat\Sigma$ of genus 3. The Prym period $\tau_{h,g}$ is related to the period matrix $\Omega$ by the Schottky-Jung relation [106, (1.6)]

$$\left(\frac{\vartheta_i(0,\tau_{h,g})}{\vartheta_j(0,\tau_{h,g})}\right)^4 = \left(\frac{\vartheta^{(2)}[\delta_i^+](0,\Omega)\, \vartheta^{(2)}[\delta_i^-](0,\Omega)}{\vartheta^{(2)}[\delta_j^+](0,\Omega)\, \vartheta^{(2)}[\delta_j^-](0,\Omega)}\right)^2 \qquad (B.31)$$

for any choice of distinct $i, j \in \{1, 2, 3\}$. Here, $\delta_i^\pm$ are the 6 even spin structures $\delta$ such that $\delta + \frac{1}{2}\begin{bmatrix} h \\ g \end{bmatrix}$ is also en even spin structure; moreover $\delta_i^- = \delta_i^+ + \frac{1}{2}\begin{bmatrix} h \\ g \end{bmatrix}$. The relation (B.31) ensures that (B.30) is independent of the choice of $i$. Since all 15 non-trivial twists are permuted by $Sp(4,\mathbb{Z})$, it will be convenient to focus on the twisted sector $\begin{bmatrix} h \\ g \end{bmatrix} = \begin{bmatrix} 00 \\ 01 \end{bmatrix}$, in which case the relation (B.31) becomes [106, (6.5)]

$$\frac{\vartheta_4^4(\tau)}{\vartheta_2^4(\tau)} = \left(\frac{\vartheta^{(2)}\begin{bmatrix} 01 \\ 00 \end{bmatrix}\vartheta^{(2)}\begin{bmatrix} 01 \\ 01 \end{bmatrix}}{\vartheta^{(2)}\begin{bmatrix} 10 \\ 00 \end{bmatrix}\vartheta^{(2)}\begin{bmatrix} 10 \\ 01 \end{bmatrix}}\right)^2 \,, \qquad (B.32)$$

where $\tau \equiv \tau_{h,g}$. In particular, under $(\rho,\sigma,\nu) \to (\rho+1,\sigma,\nu)$, the Prym period transforms as $\tau \to \tau + 1$, whereas in the non-separating degeneration $\sigma \to i\infty$, $\tau \sim \rho \bmod 4\mathbb{Z}$ [106, §7.2].

In our case, we need the orbifold blocks of 16 chiral scalars under exchange $X_i \mapsto X_{i+8 \bmod 16}$. By decomposing $X_i$ into its even and odd components $X_i \pm X_{i+8 \bmod 16}$, we find that the orbifold blocks are given by

$$\frac{\left[\vartheta^{(2)}[\delta_i^+](\Omega)\, \vartheta^{(2)}[\delta_i^-](\Omega)\right]^4}{Z_0^{16}\, \vartheta_i(\tau_{h,g})^8} \times \sum_{\mathcal{P} \in (\Lambda_{E_8}/2\Lambda_{E_8})} \Theta_{E_8[2],(\mathcal{P},0)}^{(2)}(\Omega)\, \Theta_{E_8[2],\mathcal{P}}(\tau_{h,g}) \,. \qquad (B.33)$$

As a consistency check on this result (first obtained in [11] from the partition function of the $E_8$ root lattice on the genus 3 covering surface $\hat\Sigma$), let us consider the maximal non-separating degeneration limit: the imaginary part of the period matrix $\Omega_2 = \begin{pmatrix} L_1 + L_2 & L_2 \\ L_2 & L_2 + L_3 \end{pmatrix}$ parametrizes Schwinger times along the three edges of the two-loop sunset diagram shown in Figure 1

iii). Assuming that the $\mathbb{Z}_2$ action is inserted along the edge of length $L_3$, the $E_8 \oplus E_8$ momenta running in the three edges are $(p_1, p_2), (p_1 + q, p_2 + q), (q, q)$. Decomposing as usual $p_1 + p_2 = 2\Sigma + \mathcal{P}, p_1 - p_2 = 2\Delta - \mathcal{P}$, the classical action is

$$
\begin{aligned}
& L_1(p_1^2 + p_2^2) + L_2 \left[ (p_1 + q)^2 + (p_2 + q)^2 \right] + 2L_3 q^2 \\
&= 2(L_1 + L_2) \left[ \left( \Sigma + \tfrac{1}{2}\mathcal{P} \right)^2 + \left( \Delta - \tfrac{1}{2}\mathcal{P} \right)^2 \right] + 2(L_2 + L_3) q^2 + 4L_2 (\Sigma + \tfrac{1}{2}\mathcal{P}) \cdot q \\
&= 2 \begin{pmatrix} \Sigma + \tfrac{1}{2}\mathcal{P} & q \end{pmatrix} \cdot \begin{pmatrix} L_1 + L_2 & L_2 \\ L_2 & L_2 + L_3 \end{pmatrix} \cdot \begin{pmatrix} \Sigma + \tfrac{1}{2}\mathcal{P} \\ q \end{pmatrix} + 2(L_1 + L_2) \left( \Delta - \tfrac{1}{2}\mathcal{P} \right)^2 ,
\end{aligned}
\tag{B.34}
$$

in agreement with the maximal non-separating degeneration limit of the second factor in (B.33), using $\tau_{h,g} \sim \rho$.

The contributions of the other degrees of freedom (spacetime bosons and fermions, ghosts) are unaffected by the orbifolding and, as in the maximal rank, turn the factor $1/Z_0^{16}$ in (B.33) into $1/\Phi_{10}$. In the sector $\left[ \begin{smallmatrix} h \\ g \end{smallmatrix} \right] = \left[ \begin{smallmatrix} 00 \\ 01 \end{smallmatrix} \right]$, the resulting ratio can be written in three equivalent ways [11, (4.29-31)],

$$
\begin{aligned}
\frac{\left[ \vartheta^{(2)}[\delta_i^+](\Omega) \vartheta^{(2)}[\delta_i^-](\Omega) \right]^4}{\vartheta_i(\tau_{h,g})^8 \Phi_{10}(\Omega)} &= \frac{\vartheta^{(2)} \left[ \begin{smallmatrix} 00 \\ 00 \end{smallmatrix} \right]^2 \vartheta^{(2)} \left[ \begin{smallmatrix} 00 \\ 01 \end{smallmatrix} \right]^2 \vartheta^{(2)} \left[ \begin{smallmatrix} 00 \\ 10 \end{smallmatrix} \right]^2 \vartheta^{(2)} \left[ \begin{smallmatrix} 00 \\ 11 \end{smallmatrix} \right]^2}{\vartheta_3^4 \vartheta_4^4(\tau) \Phi_{10}(\Omega)} = \frac{1}{\vartheta_3^4 \vartheta_4^4(\tau) \Phi_{6,0}(\Omega)} \\
&= \frac{\vartheta^{(2)} \left[ \begin{smallmatrix} 00 \\ 00 \end{smallmatrix} \right]^2 \vartheta^{(2)} \left[ \begin{smallmatrix} 00 \\ 01 \end{smallmatrix} \right]^2 \vartheta^{(2)} \left[ \begin{smallmatrix} 10 \\ 00 \end{smallmatrix} \right]^2 \vartheta^{(2)} \left[ \begin{smallmatrix} 10 \\ 01 \end{smallmatrix} \right]^2}{\vartheta_3^4 \vartheta_2^4(\tau) \Phi_{10}(\Omega)} = \frac{1}{\vartheta_3^4 \vartheta_2^4(\tau) \Phi_{6,1}(\Omega)} \\
&= \frac{\vartheta^{(2)} \left[ \begin{smallmatrix} 10 \\ 00 \end{smallmatrix} \right]^2 \vartheta^{(2)} \left[ \begin{smallmatrix} 10 \\ 01 \end{smallmatrix} \right]^2 \vartheta^{(2)} \left[ \begin{smallmatrix} 00 \\ 10 \end{smallmatrix} \right]^2 \vartheta^{(2)} \left[ \begin{smallmatrix} 00 \\ 11 \end{smallmatrix} \right]^2}{\vartheta_3^4 \vartheta_4^4(\tau) \Phi_{10}(\Omega)} = \frac{1}{\vartheta_2^4 \vartheta_4^4(\tau) \Phi_{6,2}(\Omega)} ,
\end{aligned}
\tag{B.35}
$$

where $\Phi_{6,0} \equiv \Phi_6$ is the Siegel modular form (A.46) of weight 6 and level 2, and $\Phi_{6,1} \propto \tilde{\Phi}_6$ and $\Phi_{6,2}$ are its images under $S_\rho$ and $T_\rho \cdot S_\rho$, respectively (see (B.43) below). Using the identity

$$
\sum_{\mathcal{P} \in (\Lambda_{E_8}/2\Lambda_{E_8})} \Theta^{(2)}_{E_8[2],(\mathcal{P},0)}(\Omega) \Theta_{E_8[2],\mathcal{P}}(\tau) =
$$
$$
\vartheta_3^4 \vartheta_4^4 \Theta^{(2)}_{E_8}(2\rho, 2\sigma, 2v) + \frac{1}{16} \vartheta_3^4 \vartheta_2^4 \Theta^{(2)}_{E_8}(\tfrac{\rho}{2}, 2\sigma, v) + \frac{1}{16} \vartheta_2^4 \vartheta_4^4 \Theta^{(2)}_{E_8}(\tfrac{\rho+1}{2}, 2\sigma, v),
\tag{B.36}
$$

we find that the orbifold block in the sector $\left[ \begin{smallmatrix} h \\ g \end{smallmatrix} \right] = \left[ \begin{smallmatrix} 00 \\ 01 \end{smallmatrix} \right]$ is given by [11, (4.38)]

$$
Z_8^{(2)} \left[ \begin{smallmatrix} 00 \\ 01 \end{smallmatrix} \right] = \frac{\Theta^{(2)}_{E_8}(2\rho, 2\sigma, 2v)}{\Phi_{6,0}} + \frac{\Theta^{(2)}_{E_8}(\tfrac{\rho}{2}, 2\sigma, v)}{16\Phi_{6,1}} + \frac{\Theta^{(2)}_{E_8}(\tfrac{\rho+1}{2}, 2\sigma, v)}{16\Phi_{6,2}} .
\tag{B.37}
$$

In particular, the dependence on the Prym period $\tau$ has disappeared. The result (B.37) is invariant under the index 15 subgroup $\Gamma_{2,e_1}(2)$ of $Sp(4, \mathbb{Z})$ which preserves the twist $\left[ \begin{smallmatrix} 00 \\ 01 \end{smallmatrix} \right]$ [106, §6.1]. In fact it can be rewritten as

$$
Z_8^{(2)} \left[ \begin{smallmatrix} 00 \\ 01 \end{smallmatrix} \right] = \sum_{\gamma \in \Gamma_{2,e_1}(2)/\Gamma_{2,0,e_1}(2)} \left[ \frac{\Theta^{(2)}_{E_8}(2\rho, 2\sigma, 2v)}{\Phi_{6,0}} \right]\Big|_\gamma ,
\tag{B.38}
$$

where $\Gamma_{2,0,e_1}(2) \equiv \Gamma_{2,e_1}(2) \cap \Gamma_{2,0}(2)$ has index 3 inside $\Gamma_{2,e_1}(2)$, and 3 inside $\Gamma_{2,0}(2)$. As a consistency check in (B.37), in the separating degeneration limit $v \to 0$ (B.37) becomes

$$
Z_8^{(2)} \left[ \begin{smallmatrix} 00 \\ 01 \end{smallmatrix} \right] \sim -4\pi^2 v^2 \frac{E_4(2\sigma)}{\eta \left[ \begin{smallmatrix} 0 \\ 1 \end{smallmatrix} \right](\sigma)} \left[ \frac{E_4(2\rho)}{\eta \left[ \begin{smallmatrix} 0 \\ 1 \end{smallmatrix} \right](\rho)} + \frac{E_4(\tfrac{\rho}{2})}{\eta \left[ \begin{smallmatrix} 1 \\ 0 \end{smallmatrix} \right](\rho)} + \frac{E_4(\tfrac{\rho+1}{2})}{\eta \left[ \begin{smallmatrix} 1 \\ 1 \end{smallmatrix} \right](\rho)} \right],
\tag{B.39}
$$

Table 1: List of genus-two orbifold blocks for the $\mathbb{Z}_2$ CHL model

| $\begin{bmatrix} h_1 h_2 \\ g_1 g_2 \end{bmatrix}$ | $Z_8^{(2)}\begin{bmatrix} h_1 h_2 \\ g_1 g_2 \end{bmatrix}$ | $\gamma \in Sp(4,\mathbb{Z})/\Gamma_{2,e_1}(2)$ |
|---|---|---|
| $\begin{bmatrix} 00 \\ 10 \end{bmatrix}$ | $\dfrac{\Theta^{(2)}_{E_8}(2\rho,2\sigma,2v)}{\Phi_{6,0}} + \dfrac{\Theta^{(2)}_{E_8}(2\rho,\frac{\sigma}{2},v)}{2^4\Phi_{6,3}} + \dfrac{\Theta^{(2)}_{E_8}(2\rho,\frac{\sigma+1}{2},v)}{2^4\Phi_{6,4}}$ | $\begin{pmatrix} 0 & 1 & 0 & 0 \\ 1 & 0 & 0 & 0 \\ 0 & 0 & 0 & 1 \\ 0 & 0 & 1 & 0 \end{pmatrix}$ |
| $\begin{bmatrix} 01 \\ 00 \end{bmatrix}$ | $\dfrac{\Theta^{(2)}_{E_8}(2\rho,\frac{\sigma}{2},v)}{2^4\Phi_{6,3}} + \dfrac{\Theta^{(2)}_{E_8}(\frac{\rho}{2},\frac{\sigma}{2},\frac{v}{2})}{2^8\Phi_{6,5}} + \dfrac{\Theta^{(2)}_{E_8}(\frac{\rho+1}{2},\frac{\sigma}{2},\frac{v}{2})}{2^8\Phi_{6,6}}$ | $\begin{pmatrix} 1 & 0 & 0 & 0 \\ 0 & 0 & 0 & -1 \\ 0 & 0 & 1 & 0 \\ 0 & 1 & 0 & 0 \end{pmatrix}$ |
| $\begin{bmatrix} 10 \\ 00 \end{bmatrix}$ | $\dfrac{\Theta^{(2)}_{E_8}(\frac{\rho}{2},2\sigma,v)}{2^4\Phi_{6,1}} + \dfrac{\Theta^{(2)}_{E_8}(\frac{\rho}{2},\frac{\sigma}{2},\frac{v}{2})}{2^8\Phi_{6,5}} + \dfrac{\Theta^{(2)}_{E_8}(\frac{\rho}{2},\frac{\sigma+1}{2},\frac{v}{2})}{2^8\Phi_{6,7}}$ | $\begin{pmatrix} 0 & 0 & 0 & -1 \\ 1 & 0 & 0 & 0 \\ 0 & 1 & 0 & 0 \\ 0 & 0 & 1 & 0 \end{pmatrix}$ |
| $\begin{bmatrix} 11 \\ 00 \end{bmatrix}$ | $\dfrac{\Theta^{(2)}_{E_8}(\frac{\rho}{2},\frac{\sigma}{2},\frac{v}{2})}{2^8\Phi_{6,5}} + \dfrac{\Theta^{(2)}_{E_8}(\frac{\rho+1}{2},\frac{\sigma+1}{2},\frac{v+1}{2})}{2^8\Phi_{6,9}} + \dfrac{\Theta^{(2)}_{E_8}(2\rho,\frac{\sigma-2v+\rho}{2},v-\rho)}{2^4\Phi_{6,13}}$ | $\begin{pmatrix} 1 & 0 & 0 & 0 \\ -1 & 0 & 0 & -1 \\ 0 & 1 & 1 & 0 \\ 0 & 1 & 0 & 0 \end{pmatrix}$ |
| $\begin{bmatrix} 01 \\ 01 \end{bmatrix}$ | $\dfrac{\Theta^{(2)}_{E_8}(2\rho,\frac{\sigma+1}{2},v)}{2^4\Phi_{6,4}} + \dfrac{\Theta^{(2)}_{E_8}(\frac{\rho}{2},\frac{\sigma+1}{2},\frac{v}{2})}{2^8\Phi_{6,7}} + \dfrac{\Theta^{(2)}_{E_8}(\frac{\rho+1}{2},\frac{\sigma+1}{2},\frac{v}{2})}{2^8\Phi_{6,8}}$ | $\begin{pmatrix} 1 & 0 & 0 & 0 \\ 0 & 1 & 0 & -1 \\ 0 & 0 & 1 & 0 \\ 0 & 1 & 0 & 0 \end{pmatrix}$ |
| $\begin{bmatrix} 10 \\ 10 \end{bmatrix}$ | $\dfrac{\Theta^{(2)}_{E_8}(\frac{\rho+1}{2},2\sigma,v)}{2^4\Phi_{6,2}} + \dfrac{\Theta^{(2)}_{E_8}(\frac{\rho+1}{2},\frac{\sigma}{2},\frac{v}{2})}{2^8\Phi_{6,6}} + \dfrac{\Theta^{(2)}_{E_8}(\frac{\rho+1}{2},\frac{\sigma+1}{2},\frac{v}{2})}{2^8\Phi_{6,8}}$ | $\begin{pmatrix} 0 & 1 & 0 & -1 \\ 1 & 0 & 0 & 0 \\ 0 & 1 & 0 & 0 \\ 0 & 0 & 1 & 0 \end{pmatrix}$ |
| $\begin{bmatrix} 01 \\ 10 \end{bmatrix}$ | $\dfrac{\Theta^{(2)}_{E_8}(2\rho,\frac{\sigma}{2},v)}{2^4\Phi_{6,3}} + \dfrac{\Theta^{(2)}_{E_8}(\frac{\rho}{2},\frac{\sigma}{2},\frac{v+1}{2})}{2^8\Phi_{6,10}} + \dfrac{\Theta^{(2)}_{E_8}(\frac{\rho+1}{2},\frac{\sigma}{2},\frac{v+1}{2})}{2^8\Phi_{6,11}}$ | $\begin{pmatrix} 0 & 1 & 0 & 0 \\ 0 & 0 & -1 & 0 \\ 0 & 0 & 1 & 1 \\ 1 & -1 & 0 & 0 \end{pmatrix}$ |
| $\begin{bmatrix} 10 \\ 11 \end{bmatrix}$ | $\dfrac{\Theta^{(2)}_{E_8}(\frac{\rho+1}{2},2\sigma,v)}{2^4\Phi_{6,2}} + \dfrac{\Theta^{(2)}_{E_8}(\frac{\rho+1}{2},\frac{\sigma+1}{2},\frac{v+1}{2})}{2^8\Phi_{6,9}} + \dfrac{\Theta^{(2)}_{E_8}(\frac{\rho+1}{2},\frac{\sigma}{2},\frac{v+1}{2})}{2^8\Phi_{6,11}}$ | $\begin{pmatrix} 0 & 1 & -1 & -1 \\ 1 & -1 & 0 & 0 \\ 0 & 1 & 0 & 0 \\ 0 & 0 & 1 & 0 \end{pmatrix}$ |
| $\begin{bmatrix} 10 \\ 01 \end{bmatrix}$ | $\dfrac{\Theta^{(2)}_{E_8}(\frac{\rho}{2},2\sigma,v)}{2^4\Phi_{6,1}} + \dfrac{\Theta^{(2)}_{E_8}(\frac{\rho}{2},\frac{\sigma}{2},\frac{v+1}{2})}{2^8\Phi_{6,10}} + \dfrac{\Theta^{(2)}_{E_8}(\frac{\rho}{2},\frac{\sigma+1}{2},\frac{v+1}{2})}{2^8\Phi_{6,12}}$ | $\begin{pmatrix} 0 & 0 & -1 & -1 \\ 1 & -1 & 0 & 0 \\ 0 & 1 & 0 & 0 \\ 0 & 0 & 1 & 0 \end{pmatrix}$ |
| $\begin{bmatrix} 01 \\ 11 \end{bmatrix}$ | $\dfrac{\Theta^{(2)}_{E_8}(2\rho,\frac{\sigma+1}{2},v)}{2^4\Phi_{6,4}} + \dfrac{\Theta^{(2)}_{E_8}(\frac{\rho+1}{2},\frac{\sigma+1}{2},\frac{v+1}{2})}{2^8\Phi_{6,9}} + \dfrac{\Theta^{(2)}_{E_8}(\frac{\rho}{2},\frac{\sigma+1}{2},\frac{v+1}{2})}{2^8\Phi_{6,12}}$ | $\begin{pmatrix} 0 & 1 & 0 & 0 \\ 1 & -1 & -1 & 0 \\ 0 & 0 & 1 & 1 \\ 1 & -1 & 0 & 0 \end{pmatrix}$ |
| $\begin{bmatrix} 00 \\ 11 \end{bmatrix}$ | $\dfrac{\Theta^{(2)}_{E_8}(2\rho,2\sigma,2v)}{\Phi_{6,0}} + \dfrac{\Theta^{(2)}_{E_8}(2\rho,\frac{\rho-2v+\sigma}{2},v-\rho)}{2^4\Phi_{6,13}} + \dfrac{\Theta^{(2)}_{E_8}(2\rho,\frac{\rho-2v+\sigma+1}{2},v-\rho)}{2^4\Phi_{6,14}}$ | $\begin{pmatrix} 0 & 1 & 0 & 0 \\ 1 & -1 & 0 & 0 \\ 0 & 0 & 1 & 1 \\ 0 & 0 & 1 & 0 \end{pmatrix}$ |
| $\begin{bmatrix} 11 \\ 01 \end{bmatrix}$ | $\dfrac{\Theta^{(2)}_{E_8}(\frac{\rho}{2},\frac{\sigma+1}{2},\frac{v}{2})}{2^8\Phi_{6,7}} + \dfrac{\Theta^{(2)}_{E_8}(\frac{\rho+1}{2},\frac{\sigma}{2},\frac{v+1}{2})}{2^8\Phi_{6,11}} + \dfrac{\Theta^{(2)}_{E_8}(2\rho,\frac{\rho-2v+\sigma+1}{2},v-\rho)}{2^4\Phi_{6,14}}$ | $\begin{pmatrix} 1 & 0 & 0 & 0 \\ -1 & 1 & 0 & -1 \\ 0 & 1 & 1 & 0 \\ 0 & 1 & 0 & 0 \end{pmatrix}$ |
| $\begin{bmatrix} 11 \\ 10 \end{bmatrix}$ | $\dfrac{\Theta^{(2)}_{E_8}(\frac{\rho+1}{2},\frac{\sigma}{2},\frac{v}{2})}{2^8\Phi_{6,6}} + \dfrac{\Theta^{(2)}_{E_8}(\frac{\rho}{2},\frac{\sigma+1}{2},\frac{v+1}{2})}{2^8\Phi_{6,12}} + \dfrac{\Theta^{(2)}_{E_8}(2\rho,\frac{\rho-2v+\sigma+1}{2},v-\rho)}{2^4\Phi_{6,14}}$ | $\begin{pmatrix} 1 & 1 & 1 & 0 \\ -1 & 0 & 0 & -1 \\ 0 & 1 & 1 & 0 \\ 0 & 1 & 0 & 0 \end{pmatrix}$ |
| $\begin{bmatrix} 11 \\ 11 \end{bmatrix}$ | $\dfrac{\Theta^{(2)}_{E_8}(\frac{\rho+1}{2},\frac{\sigma+1}{2},\frac{v}{2})}{2^8\Phi_{6,8}} + \dfrac{\Theta^{(2)}_{E_8}(\frac{\rho}{2},\frac{\sigma}{2},\frac{v+1}{2})}{2^8\Phi_{6,10}} + \dfrac{\Theta^{(2)}_{E_8}(2\rho,\frac{\sigma-2v+\rho}{2},v-\rho)}{2^4\Phi_{6,13}}$ | $\begin{pmatrix} 1 & 1 & 1 & 0 \\ -1 & 1 & 0 & -1 \\ 0 & 1 & 1 & 0 \\ 0 & 1 & 0 & 0 \end{pmatrix}$ |

where, for $N$ prime and $h \neq 0 \bmod N$ we define

$$\eta\begin{bmatrix} 0 \\ g \end{bmatrix} = \eta^{k+2}(\tau)\,\eta^{k+2}(N\tau)\,, \qquad \eta\begin{bmatrix} h \\ g \end{bmatrix} = e^{\frac{i\pi a(k+2)}{12}}\,\eta^{k+2}(\tau)\,\eta^{k+2}\left(\frac{\tau+a}{N}\right), \tag{B.40}$$

where $k+2 = \ell$, $a = gh^{-1}$, with $h^{-1}$ being the inverse of $h$ in the multiplicative group $\mathbb{Z}/N\mathbb{Z}$.

Using [22, Eq.(A.10)], the term in bracket is indeed recognized as the untwisted unprojected one-loop partition function

$$Z_8^{(2)}\begin{bmatrix} 0 \\ 0 \end{bmatrix} \equiv \frac{E_4^2}{\eta^{24}} = \frac{E_4(2\tau)}{\eta^8(\tau)\eta^8(2\tau)} + \frac{E_4(\frac{\tau}{2})}{\eta^8(\tau)\eta^8(\frac{\tau}{2})} + \frac{E_4(\frac{\tau+1}{2})}{e^{2i\pi/3}\eta^8(\tau)\eta^8(\frac{\tau+1}{2})} \ . \tag{B.41}$$

The remaining blocks can be obtained by modular transformations,

$$Z_8^{(2)}[\tilde{\delta}](\tilde{\Omega}) = Z_8[\delta](\Omega) \ , \quad \tilde{\delta} = \begin{pmatrix} D & -C \\ -B & A \end{pmatrix} \delta \bmod 2 \ , \tag{B.42}$$

where $\delta = (h_1, h_2, g_1, g_2)^t$. Using the invariance of $\Theta_{E_8}^{(2)}(\rho, \sigma, \nu)$ under the full Siegel modular group, and acting with the 15 elements $\gamma$ of $Sp(4, \mathbb{Z})/\Gamma_{2,e_1}(2)$ on (B.38), we obtain the orbifold blocks shown on Table 1. In this table, $\Phi_{6,1}$ through $\Phi_{6,14}$ are images of $\Phi_{6,0}$ under $\gamma \in Sp(4, \mathbb{Z})/\Gamma_{2,0}(2)$. When $\gamma$ lies in $SL(2,\mathbb{Z})_\rho \times SL(2,\mathbb{Z})_\sigma \to Sp(4,\mathbb{Z})$ we denote the respective $SL(2,\mathbb{Z})$ generators in subscript:

$$\Phi_{6,1}(\rho,\sigma,\nu) = \rho^{-6}\Phi_{6,0}(-1/\rho, \sigma - \nu^2/\rho, \nu/\rho) = \Phi_6\Big|\begin{pmatrix} 0 & 0 & -1 & 0 \\ 0 & 1 & 0 & 0 \\ 1 & 0 & 0 & 0 \\ 0 & 0 & 0 & 1 \end{pmatrix} = \Phi_6|_{(S,\mathbb{1})} \ ,$$

$$\Phi_{6,2}(\rho,\sigma,\nu) = \Phi_{6,1}(\rho+1, \sigma, \nu) = \Phi_6\Big|\begin{pmatrix} 1 & 0 & -1 & 0 \\ 0 & 1 & 0 & 0 \\ 1 & 0 & 0 & 0 \\ 0 & 0 & 0 & 1 \end{pmatrix} = \Phi_6|_{(TS,\mathbb{1})} \ ,$$

$$\Phi_{6,3}(\rho,\sigma,\nu) = \sigma^{-6}\Phi_{6,0}(\rho - \nu^2/\sigma, -1/\sigma, \nu/\sigma) = \Phi_6\Big|\begin{pmatrix} 1 & 0 & 0 & 0 \\ 0 & 0 & 0 & -1 \\ 0 & 0 & 1 & 0 \\ 0 & 1 & 0 & 0 \end{pmatrix} = \Phi_6|_{(\mathbb{1},S)} \ ,$$

$$\Phi_{6,4}(\rho,\sigma,\nu) = \Phi_{6,3}(\rho, \sigma+1, \nu) = \Phi_6\Big|\begin{pmatrix} 1 & 0 & 0 & 0 \\ 0 & 1 & 0 & -1 \\ 0 & 0 & 1 & 0 \\ 0 & 1 & 0 & 0 \end{pmatrix} = \Phi_6|_{(\mathbb{1},TS)} \ , \tag{B.43}$$

$$\Phi_{6,5}(\rho,\sigma,\nu) = \sigma^{-6}\Phi_{6,1}(\rho - \nu^2/\sigma, -1/\sigma, \nu/\sigma) = \Phi_6\Big|\begin{pmatrix} 0 & 0 & -1 & 0 \\ 0 & 0 & 0 & -1 \\ 1 & 0 & 0 & 0 \\ 0 & 1 & 0 & 0 \end{pmatrix} = \Phi_6|_{(S,S)} \ ,$$

$$\Phi_{6,6}(\rho,\sigma,\nu) = \Phi_{6,5}(\rho+1, \sigma, \nu) = \Phi_6\Big|\begin{pmatrix} 1 & 0 & -1 & 0 \\ 0 & 0 & 0 & -1 \\ 1 & 0 & 0 & 0 \\ 0 & 1 & 0 & 0 \end{pmatrix} = \Phi_6|_{(TS,S)} \ ,$$

$$\Phi_{6,7}(\rho,\sigma,\nu) = \Phi_{6,5}(\rho, \sigma+1, \nu) = \Phi_6\Big|\begin{pmatrix} 0 & 0 & -1 & 0 \\ 0 & 1 & 0 & -1 \\ 1 & 0 & 0 & 0 \\ 0 & 1 & 0 & 0 \end{pmatrix} = \Phi_6|_{(S,TS)} \ ,$$

$$\Phi_{6,8}(\rho,\sigma,\nu) = \Phi_{6,5}(\rho+1, \sigma+1, \nu) = \Phi_6\Big|\begin{pmatrix} 1 & 0 & -1 & 0 \\ 0 & 1 & 0 & -1 \\ 1 & 0 & 0 & 0 \\ 0 & 1 & 0 & 0 \end{pmatrix} = \Phi_6|_{(TS,TS)} \ ,$$

$$\Phi_{6,9}(\rho,\sigma,v) = \Phi_{6,5}(\rho+1,\sigma+1,v+1) = \Phi_6\Big|\begin{pmatrix} 1 & 1 & -1 & 0 \\ 1 & 1 & 0 & -1 \\ 1 & 0 & 0 & 0 \\ 0 & 1 & 0 & 0 \end{pmatrix},$$

$$\Phi_{6,10}(\rho,\sigma,v) = \Phi_{6,5}(\rho,\sigma,v+1) = \Phi_6\Big|\begin{pmatrix} 0 & 1 & -1 & 0 \\ 1 & 0 & 0 & -1 \\ 1 & 0 & 0 & 0 \\ 0 & 1 & 0 & 0 \end{pmatrix},$$

$$\Phi_{6,11}(\rho,\sigma,v) = \Phi_{6,5}(\rho+1,\sigma,v+1) = \Phi_6\Big|\begin{pmatrix} 1 & 1 & -1 & 0 \\ 1 & 0 & 0 & -1 \\ 1 & 0 & 0 & 0 \\ 0 & 1 & 0 & 0 \end{pmatrix}, \tag{B.44}$$

$$\Phi_{6,12}(\rho,\sigma,v) = \Phi_{6,5}(\rho,\sigma+1,v+1) = \Phi_6\Big|\begin{pmatrix} 0 & 1 & -1 & 0 \\ 1 & 1 & 0 & -1 \\ 1 & 0 & 0 & 0 \\ 0 & 1 & 0 & 0 \end{pmatrix},$$

$$\Phi_{6,13}(\rho,\sigma,v) = \Phi_{6,3}(\rho,\sigma-2v+\rho,v-\rho) = \Phi_6\Big|\begin{pmatrix} 1 & 0 & 0 & 0 \\ -1 & 0 & 0 & -1 \\ 0 & 1 & 1 & 0 \\ 0 & 1 & 0 & 0 \end{pmatrix},$$

$$\Phi_{6,14}(\rho,\sigma,v) = \Phi_{6,4}(\rho,\sigma-2v+\rho+1,v-\rho) = \Phi_6\Big|\begin{pmatrix} 1 & 0 & 0 & 0 \\ -1 & 1 & 0 & -1 \\ 0 & 1 & 1 & 0 \\ 0 & 1 & 0 & 0 \end{pmatrix}.$$

As a consistency check, using the fact that

$$\Phi_{6,k}(\rho,\sigma,v) \sim \begin{cases} 2^{-8}v^2\eta^{12}\vartheta_i^4(\rho)\vartheta_j^4(\sigma) + \mathcal{O}(v^4), & k \leq 8 \\ \pm 2^{-4}\eta^{12}(\rho)\eta^{12}(\sigma) + \mathcal{O}(v^2) & k \geq 9 \end{cases}, \tag{B.45}$$

where $(k,i,j) = (0,2,2),(1,4,2),(2,3,2)(3,2,4),(4,2,3),(5,4,4),(6,3,4),(7,4,3),(8,3,3)$ for $k \leq 8$, we see that in the separating degeneration limit $v \to 0$,

$$Z_8^{(2)}\big[{}^{h_1 h_2}_{g_1 g_2}\big](\Omega) \sim -4\pi^2 v^2 Z_8^{(1)}\big[{}^{h_1}_{g_1}\big](\rho) Z_8^{(1)}\big[{}^{h_2}_{g_2}\big](\sigma) + \mathcal{O}(v^2), \tag{B.46}$$

where $Z_8^{(1)}\big[{}^{h}_{g}\big]$ are the genus-one orbifold blocks given in [22, Eq.(A.6)]. Note that each of the numerators appearing in the genus-two orbifold blocks $Z_8^{(2)}\big[{}^{h_1 h_2}_{g_1 g_2}\big]$ can be interpreted as the genus-two theta series for an Euclidean lattice of rank 8 as follows (here $q^{\frac{1}{2}Q^2}$ denotes $e^{i\pi Q^r \Omega_{rs} Q^s}$)

$$\Theta_{E_8}(2\rho,2v,2\sigma) = \sum_{\substack{(Q_1,Q_2) \in \\ E_8[2] \oplus E_8[2]}} e^{i\pi Q^r \Omega_{rs} Q^s},$$

$$\Theta_{E_8}(2\rho,v,\tfrac{\sigma}{2}) = 2^{-4} \sum_{\substack{(Q_1,Q_2) \in \\ E_8[2] \oplus E_8[2]^*}} e^{i\pi Q^r \Omega_{rs} Q^s},$$

$$\Theta_{E_8}(2\rho,v,\tfrac{\sigma+1}{2}) = 2^{-4} \sum_{\substack{(Q_1,Q_2) \in \\ E_8[2] \oplus E_8[2]^*}} (-1)^{Q_2^2} e^{i\pi Q^r \Omega_{rs} Q^s}, \tag{B.47}$$

$$\Theta_{E_8}(\tfrac{\rho}{2},\tfrac{v}{2},\tfrac{\sigma+1}{2}) = 2^{-8} \sum_{\substack{(Q_1,Q_2) \in \\ E_8[2]^* \oplus E_8[2]^*}} (-1)^{Q_2^2} e^{i\pi Q^r \Omega_{rs} Q^s},$$

$$\Theta_{E_8}\left(\tfrac{\rho}{2}, \tfrac{\nu+1}{2}, \tfrac{\sigma}{2}\right) = 2^{-8} \sum_{\substack{(Q_1,Q_2)\in \\ E_8[2]^* \oplus E_8[2]^*}} (-1)^{2Q_1 \cdot Q_2} e^{i\pi Q^r \Omega_{rs} Q^s},$$

$$\Theta_{E_8}\left(\tfrac{\rho}{2}, \tfrac{\nu+1}{2}, \tfrac{\sigma+1}{2}\right) = 2^{-8} \sum_{\substack{(Q_1,Q_2)\in \\ E_8[2]^* \oplus E_8[2]^*}} (-1)^{(2Q_1+Q_2)\cdot Q_2} e^{i\pi Q^r \Omega_{rs} Q^s},$$

$$\Theta_{E_8}\left(\tfrac{\rho+1}{2}, \tfrac{\nu}{2}, \tfrac{\sigma+1}{2}\right) = 2^{-8} \sum_{\substack{(Q_1,Q_2)\in \\ E_8[2]^* \oplus E_8[2]^*}} (-1)^{Q_1^2+Q_2^2} e^{i\pi Q^r \Omega_{rs} Q^s},$$

$$\Theta_{E_8}\left(\tfrac{\rho+1}{2}, \tfrac{\nu+1}{2}, \tfrac{\sigma+1}{2}\right) = 2^{-8} \sum_{\substack{(Q_1,Q_2)\in \\ E_8[2]^* \oplus E_8[2]^*}} (-1)^{(Q_1+Q_2)^2} e^{i\pi Q^r \Omega_{rs} Q^s}, \tag{B.48}$$

$$\Theta_{E_8}\left(2\rho, \nu-\rho, \tfrac{\sigma-2\nu+\rho}{2}\right) = 2^{-4} \sum_{\substack{(Q_1,Q_2)\in \\ E_8[2]^* \oplus E_8[2]^*}} \delta_{(Q_1+Q_2)\in E_8[2]} \, e^{i\pi Q^r \Omega_{rs} Q^s},$$

$$\Theta_{E_8}\left(2\rho, \nu-\rho, \tfrac{\sigma-2\nu+\rho+1}{2}\right) = 2^{-4} \sum_{\substack{(Q_1,Q_2)\in \\ E_8[2]^* \oplus E_8[2]^*}} \delta_{(Q_1+Q_2)\in E_8[2]} (-1)^{\frac{1}{4}(Q_1-Q_2)^2} e^{i\pi Q^r \Omega_{rs} Q^s}.$$

Now, as indicated above (B.27), the orbifold blocks $Z_8^{(2)}\left[\begin{smallmatrix} h_1 h_2 \\ g_1 g_2 \end{smallmatrix}\right]$ only include the contributions from the chiral measure for the ten-dimensional string, and need to be supplemented with the contribution of the bosonic zero-modes of the $d$ compact bosons,

$$Z_{d,d}^{(2)}\left[\begin{smallmatrix} h_1 h_2 \\ g_1 g_2 \end{smallmatrix}\right] = |\Omega_2|^{d/2} \sum_{Q\in \Lambda_{d,d}^{\otimes 2}+\frac{1}{2}(h_1,h_2)\delta} (-1)^{\delta\cdot(g_1 Q_1+g_2 Q_2)} e^{i\pi Q_L^r \Omega_{rs} Q_L^s - i\pi Q_R^r \bar{\Omega}_{rs} Q_R^s}, \tag{B.49}$$

where $\delta$ is a null element in $(2\amalg_{d,d})/\amalg_{d,d}$ which depends on the orbifold action on $T^d$; we shall henceforth restrict to a half-period shift along the $d$-th circle, so that $\delta = (0^d; 0^{d-1}1)$. For this choice, the product of (B.38) and (B.49) can again be written as a sum over images under the stabilizer of the twist,

$$Z_8^{(2)}\left[\begin{smallmatrix} 00 \\ 01 \end{smallmatrix}\right] Z_{d,d}^{(2)}\left[\begin{smallmatrix} 00 \\ 01 \end{smallmatrix}\right] = \sum_{\gamma\in\Gamma_{2,e_1}(2)/\Gamma_{2,0,e_1}(2)} \frac{\Gamma_{\tilde{\Lambda}_{d+8,d}}^{(2)}\left[(-1)^{\delta\cdot Q_2}\right]}{\Phi_{6,0}}\bigg|_{\gamma}, \tag{B.50}$$

where

$$\tilde{\Lambda}_{d+8,d} \equiv E_8[2] \oplus \amalg_{d,d}, \tag{B.51}$$

and $\delta \cdot Q_2$ equals the winding of the $d$-th embedding coordinate along the cycle $B_2$. Thus, the sum over all the sectors listed in (1), in the case of compactification on $T^d$ at this specific factorization point in the moduli space, can be rewritten as

$$\sideset{}{'}\sum_{h_r,g_r\in\{0,1\}} Z_8^{(2)}\left[\begin{smallmatrix} h_1 h_2 \\ g_1 g_2 \end{smallmatrix}\right] Z_{d,d}^{(2)}\left[\begin{smallmatrix} h_1 h_2 \\ g_1 g_2 \end{smallmatrix}\right] = \sum_{\gamma\in Sp(4,\mathbb{Z})/\Gamma_{2,e_1}(2)} Z_8^{(2)}\left[\begin{smallmatrix} 00 \\ 01 \end{smallmatrix}\right] Z_{d,d}^{(2)}\left[\begin{smallmatrix} 00 \\ 01 \end{smallmatrix}\right]\bigg|_{\gamma}$$

$$= \sum_{\gamma\in Sp(4,\mathbb{Z})/\Gamma_{2,0}(2)} \frac{\Gamma_{\tilde{\Lambda}_{d+8,d}}^{(2)}\left[\left((-1)^{\delta\cdot Q_1} + (-1)^{\delta\cdot Q_2} + (-1)^{\delta\cdot(Q_1+Q_2)}\right) P_{ab,cd}\right]}{\Phi_{6,0}}\bigg|_{\gamma}, \tag{B.52}$$

where for the last equality we expressed $Z_8^{(2)}\left[\begin{smallmatrix} 00 \\ 01 \end{smallmatrix}\right] Z_{d,d}^{(2)}\left[\begin{smallmatrix} 00 \\ 01 \end{smallmatrix}\right]$ as a sum over $\Gamma_{2,e_1}(2)/\Gamma_{2,0,e_1}(2)$, similarly to (B.38), and rewrote the two sums as a double sum over $Sp(4,\mathbb{Z})/\Gamma_{2,0}(2)$ and $\Gamma_{2,0}(2)/\Gamma_{2,0,e_1}(2)$.

Including the contribution from the second line in (B.20), and retaining the next-to-leading term in the low energy expansion, we see that the $\nabla^2 F^4$ coupling on the locus $G_{d,d} \subset G_{d+8,d}$ where the lattice $\Lambda_{d+16,d}$ factorizes is given by

$$G_{ab,cd}^{(2)} = \frac{1}{4} \, \text{R.N.} \int_{\mathcal{F}_2} \frac{d^3\Omega_1 \, d^3\Omega_2}{|\Omega_2|^3} \left( Z_8^{(2)} \begin{bmatrix} 00 \\ 00 \end{bmatrix} Z_{d,d}^{(2)} \begin{bmatrix} 00 \\ 00 \end{bmatrix} + \sum_{h_r,g_r \in \{0,1\}}^{\prime} Z_8^{(2)} \begin{bmatrix} h_1 h_2 \\ g_1 g_2 \end{bmatrix} Z_{d,d}^{(2)} \begin{bmatrix} h_1 h_2 \\ g_1 g_2 \end{bmatrix} \right) [P_{ab,cd}] \, ,$$

(B.53)

where the bracket $[P_{ab,cd}]$ denotes an insertion of the quartic polynomial $P_{ab,cd}$ (2.31) in the sum over the lattice $\tilde{\Lambda}_{d+8,d}$ and its modular images.

Now, in parallel with the 'Hecke identity' (B.41), observe that the untwisted genus-two chiral partition function satisfies

$$Z_8^{(2)} \begin{bmatrix} 00 \\ 00 \end{bmatrix} \equiv \frac{[\Theta_{E_8}^{(2)}(\Omega)]^2}{\Phi_{10}} = \sum_{h_r,g_r \in \{0,1\}}^{\prime} Z_8^{(2)} \begin{bmatrix} h_1 h_2 \\ g_1 g_2 \end{bmatrix} \, .$$

(B.54)

The validity of this identity can for example be checked for the minimal non-separating degeneration using (A.31). Using this identity in the sum over all sectors, as in (B.53), we can rewrite it as a sum over $Sp(4,\mathbb{Z})/\Gamma_{2,0}(2)$, as in the second line of (B.52), to obtain

$$\frac{1}{4} \sum_{h_r,g_r \in \{0,1\}} Z_8^{(2)} \begin{bmatrix} h_1 h_2 \\ g_1 g_2 \end{bmatrix} Z_{d,d}^{(2)} \begin{bmatrix} h_1 h_2 \\ g_1 g_2 \end{bmatrix} [P_{ab,cd}] =$$

$$\sum_{\gamma \in Sp(4,\mathbb{Z})/\Gamma_{2,0}(2)} \frac{\Gamma_{\tilde{\Lambda}_{d+8,d}}^{(2)} \left[ \frac{1}{2}\left(1+(-1)^{\delta \cdot Q_1}\right) \frac{1}{2}\left(1+(-1)^{\delta \cdot Q_2}\right) P_{ab,cd} \right]}{\Phi_{6,0}} \Bigg|_{\gamma} \, .$$

(B.55)

The insertions of $\frac{1}{2}\left(1+(-1)^{\delta \cdot Q_i}\right)$ can be seen as projectors on the lattice $\tilde{\Lambda}_{d+8,8}$ to vectors with even entries along one of the cicle designated by $\delta$, such that the resulting sum is recognized as a genus-two partition function, with insertion of $P_{ab,cd}$ only, for the 'magnetic charge lattice' introduced in [22, (A,16)],

$$\Lambda_{d+8,d} = E_8[2] \oplus I\!I_{1,1}[2] \oplus I\!I_{d-1,d-1} \, .$$

(B.56)

At this point, we can readily extend the result away from the factorized locus by allowing non-trivial Wilson lines in the lattice partition function. As established in (B.55), the partition function can be written down as a sum over images from under $Sp(4,\mathbb{Z})/\Gamma_{2,0}(2)$, such that the integral can be unfolded from a fundamental domain of $Sp(4,\mathbb{Z})$ to a fundamental domain of $\Gamma_{2,0}(2)$

$$G_{ab,cd}^{(2)} = \text{R.N.} \int_{\Gamma_{2,0}(2) \backslash \mathcal{H}_2} \frac{d^3\Omega_1 \, d^3\Omega_2}{|\Omega_2|^3} \frac{\Gamma_{\tilde{\Lambda}_{d+8,d}}^{(2)} [P_{ab,cd}]}{\Phi_6} \, .$$

(B.57)

This concludes the computation of the two-loop $\nabla^2 F^4$ coupling in the $\mathbb{Z}_2$ orbifold.

### B.2.3 $\mathbb{Z}_N$ orbifold with $N = 3, 5, 7$

Let us now briefly discuss the genus-two amplitude in heterotic CHL orbifolds with $N = 2, 3, 5, 7$. As in [22, §A.2], we restrict to a locus $G_{d+k-8,d+k-8} \subset G_{d+16,d}$ where the even self-dual lattice $\Lambda_{d+16,d}$ of the heterotic string compactified on $T^d$ factorizes as $\Lambda_{Nk,8-k} \oplus I\!I_{1,1} \oplus I\!I_{d+k-8,d+k-8}$, where the $\mathbb{Z}_N$ action acts by a $\mathbb{Z}_N$ rotation on the first factor and by a translation by $1/N$ period on the second. We denote by $\Lambda_{k,8-k}$ the $\mathbb{Z}_N$-invariant part of $\Lambda_{Nk,8-k}$, and let

$$\tilde{\Lambda}_{d+2k-8,d} = \Lambda_{k,8-k} \oplus I\!I_{1,1} \oplus I\!I_{d+k-9,d+k-9} \, .$$

(B.58)

Upon using the Niemeier lattice construction of the $\mathbb{Z}_N$-symmetric lattice outlined in [22], one finds that the invariant lattice $\Lambda_{k,8-k} = D_k[N] \oplus D_{8-k}[-1]$, where the sum is performed with respect to the diagonal glue code $\{(0,0),(s,s),(v,v),(c,c)\}$. For $N = 2$ using the construction in the previous subsection, one has instead $\Lambda_{8,0} = E_8[2]$.

Now, as in (B.26) the genus-two amplitude decomposes into a sum over all possible twisted or untwisted periodicity conditions $\begin{bmatrix} h_1 h_2 \\ g_1 g_2 \end{bmatrix}$ along the $A$ and $B$ cycles of the genus-two curve $\Sigma$, with $h_r, g_r$ running over $\mathbb{Z}/(N\mathbb{Z})$. For $N$ prime, all $N^4 - 1$ non-trivial twistings form a single orbit under $Sp(4, \mathbb{Z})$, so it suffices to focus on one of them, say $\epsilon = \begin{bmatrix} 00 \\ 01 \end{bmatrix}$. The stabilizer of $\epsilon$ under the action (B.42) is $\Gamma_{2,e_1}(N)$ (a subgroup of index $N^4 - 1$ inside $Sp(4, \mathbb{Z})$), so the corresponding orbifold block $\widetilde{Z}^{(2)}_{d+2k-8,d}\begin{bmatrix} 00 \\ 01 \end{bmatrix}$ must be a Siegel modular form for $\Gamma_{2,e_1}(N)$, and satisfy

$$\sideset{}{'}\sum_{h_r,g_r \in \mathbb{Z}/(N\mathbb{Z})} \widetilde{Z}^{(2)}_{d+2k-8,d}\begin{bmatrix} h_1 h_2 \\ g_1 g_2 \end{bmatrix} = \sum_{\gamma \in Sp(4,\mathbb{Z})/\Gamma_{2,e_1}(N)} \widetilde{Z}^{(2)}_{d+2k-8,d}\begin{bmatrix} 00 \\ 01 \end{bmatrix}\bigg|_{\gamma}. \tag{B.59}$$

This orbifold block can in principle be computed using the $N$-sheeted cover of the genus-two curve $\Sigma$, which now has genus $N + 1$. Rather than following this route, we instead postulate that it is given by the natural generalization of (B.50), namely

$$\widetilde{Z}^{(2)}_{d+2k-8,d}\begin{bmatrix} 00 \\ 01 \end{bmatrix} = \sum_{\gamma \in \Gamma_{2,e_1}(N)/\Gamma_{2,0,e_1}(N)} \frac{\Gamma^{(2)}_{\tilde{\Lambda}_{d+2k-8,d}}\left[e^{\frac{2\pi i \delta \cdot Q_2}{N}}\right]}{\Phi_{k-2}}\bigg|_{\gamma}, \tag{B.60}$$

where $\Gamma_{2,0,e_1}(N) = \Gamma_{2,e_1}(N) \cap \Gamma_{2,0}(N)$ has index $N + 1$ in $\Gamma_{2,e_1}(N)$ and $N^2 - 1$ in $\Gamma_{2,0}(N)$, and $\delta \cdot Q_2 = n_2$ is the winding of the $d$-th embedding coordinate along the cycle $B_2$, so that $\Gamma^{(2)}_{\tilde{\Lambda}_{d+2k-8,d}}\left[e^{\frac{2\pi i \delta \cdot Q_2}{N}}\right]$ is a modular form of $\Gamma_{2,0,e_1}(N)$. As a consistency check, one may verify that (B.60) has the correct behavior

$$\widetilde{Z}^{(2)}_{d+2k-8,d}\begin{bmatrix} 00 \\ 01 \end{bmatrix}(\Omega) \to -4\pi^2 v^2 \, \widetilde{Z}^{(1)}_{d+2k-8,d}\begin{bmatrix} 0 \\ 0 \end{bmatrix}(\rho) \, \widetilde{Z}^{(1)}_{d+2k-8,d}\begin{bmatrix} 0 \\ 1 \end{bmatrix}(\sigma) \tag{B.61}$$

in the separating degeneration limit $v \to 0$, where

$$\widetilde{Z}^{(1)}_{d+2k-8,d}\begin{bmatrix} 0 \\ 0 \end{bmatrix} = \sum_{\gamma \in SL(2,\mathbb{Z})/\Gamma_0(N)} \frac{\Gamma_{\tilde{\Lambda}_{d+2k-8,d}}}{\Delta_k}\bigg|_{\gamma}, \qquad \widetilde{Z}^{(1)}_{d+2k-8,d}\begin{bmatrix} 0 \\ 1 \end{bmatrix} = \frac{\Gamma_{\tilde{\Lambda}_{d+2k-8,d}}\left[e^{\frac{2\pi i \delta \cdot Q}{N}}\right]}{\Delta_k}. \tag{B.62}$$

Similarly as in the $N = 2$ case, we deduce from (B.59) and (B.60) that the sum over all non-trivial twisted sectors can be rewritten as a sum over images under $\Gamma_{2,0}(N)$,

$$\sideset{}{'}\sum_{h_i,g_i \in \mathbb{Z}/(N\mathbb{Z})} \widetilde{Z}_{d+2k-8,d}\begin{bmatrix} h_1 h_2 \\ g_1 g_2 \end{bmatrix} = \sum_{\gamma \in Sp(4,\mathbb{Z})/\Gamma_{2,e_1}(N)} \sum_{\gamma' \in \Gamma_{2,e_1}(N)/\Gamma_{2,0,e_1}(N)} \frac{\Gamma^{(2)}_{\tilde{\Lambda}_{d+2k-8,d}}\left[e^{\frac{2\pi i \delta \cdot Q_2}{N}}\right]}{\Phi_{k-2}}\bigg|_{\gamma'\gamma}$$
$$= \sum_{\gamma \in Sp(4,\mathbb{Z})/\Gamma_{2,0}(N)} \left[ \sum_{\gamma' \in \Gamma_{2,0}(N)/\Gamma_{2,0,e_1}(N)} \frac{\Gamma^{(2)}_{\tilde{\Lambda}_{d+2k-8,d}}\left[e^{\frac{2\pi i \delta \cdot Q_2}{N}}\right]}{\Phi_{k-2}}\bigg|_{\gamma'}\right]\bigg|_{\gamma}. \tag{B.63}$$

Next, we observe that the untwisted genus-two amplitude also satisfies an Hecke identity generalizing (B.54), namely

$$\widetilde{Z}_{d+2k-8,d}\begin{bmatrix} 00 \\ 00 \end{bmatrix} = \frac{\Gamma^{(2)}_{\Lambda_{d+16,d}}}{\Phi_{10}} = \sum_{\gamma \in Sp(4,\mathbb{Z})/\Gamma_{2,0}(N)} \frac{\Gamma^{(2)}_{\tilde{\Lambda}_{d+2k-8,d}}}{\Phi_{k-2}}\bigg|_{\gamma}. \tag{B.64}$$

Combining (B.60) and (B.64), and using

$$\frac{1}{N^2}\Big(\Gamma^{(2)}_{\tilde{\Lambda}_{d+2k-8,d}} + \sum_{\gamma\in\Gamma_{2,0}(N)/\Gamma_{2,0,e_1}(N)}\Gamma^{(2)}_{\tilde{\Lambda}_{d+2k-8,d}}\Big[e^{\frac{2\pi i\delta\cdot Q_2}{N}}\Big]\Big|\gamma\Big)$$
$$= \Gamma^{(2)}_{\tilde{\Lambda}_{d+2k-8,d}}\Big[\frac{1}{N}\Big(1 + e^{\frac{2\pi i\delta\cdot Q_1}{N}} + \ldots + e^{\frac{2\pi i(N-1)\delta\cdot Q_1}{N}}\Big)\frac{1}{N}\Big(1 + e^{\frac{2\pi i\delta\cdot Q_2}{N}} + \ldots + e^{\frac{2\pi i(N-1)\delta\cdot Q_2}{N}}\Big)\Big], \tag{B.65}$$

we find that the sum over all twisted sectors reduce to a sum over images under $\Gamma_{2,0}(N)$

$$\frac{1}{N^2}\sum_{h_i,g_i\in\mathbb{Z}/(N\mathbb{Z})}Z\big[{}^{h_1 h_2}_{g_1 g_2}\big] = \sum_{\gamma\in Sp(4,\mathbb{Z})/\Gamma_{2,0}(N)}\frac{\Gamma^{(2)}_{\Lambda_{d+2k-8,d}}}{\Phi_{k-2}}\Big|_\gamma, \tag{B.66}$$

where now the Siegel theta series involves the rescaled lattice

$$\Lambda_{d+2k-8,d} = \Lambda_{k,8-k} \oplus I\!I_{1,1}[N] \oplus I\!I_{d+k-9,d+k-9}. \tag{B.67}$$

After including the contribution from the second line in (B.20), retaining the next-to-leading term in the low energy expansion, and unfolding the integration domain $\mathcal{F}_2$ against the sum over images in (B.66), we conclude that the genus-two $\nabla^2 F^4$ coupling is given by

$$G^{(2)}_{ab,cd} = \text{R.N.}\int_{\Gamma_{2,0}(N)\backslash\mathcal{H}_2}\frac{d^3\Omega_1\,d^3\Omega_2}{|\Omega_2|^3}\frac{\Gamma^{(2)}_{\Lambda_{d+2k-8,d}}[P_{ab,cd}]}{\Phi_{k-2}}, \tag{B.68}$$

as announced in (2.28).

### B.2.4 Regularization of the genus-two modular integral

In order to regulate the genus-two modular integral (2.30), it is easiest to fold the integration domain $\mathcal{H}_2/\Gamma_{2,0}(N)$ back to the standard fundamental domain of $Sp(4,\mathbb{Z})$ defined in (A.5),

$$G^{(p,q)}_{ab,cd} = \text{R.N.}\int_{\mathcal{F}_2}\frac{d^3\Omega_1 d^3\Omega_2}{|\Omega_2|^3}\sum_{\gamma\in\Gamma_{2,0}(N)\backslash Sp(4,\mathbb{Z})}\frac{\Gamma^{(2)}_{\Lambda_{p,q}}[P_{ab,cd}]}{\Phi_{k-2}}\Big|_\gamma. \tag{B.69}$$

The renormalized modular integral over $\mathcal{F}_2$ can then be defined following the procedure in [67, 107], *i.e.* by truncating the fundamental domain to $\mathcal{F}_2^\Lambda = \mathcal{F}_2 \cap \{t < \Lambda\}$, where the coordinate $t$ on $\mathcal{H}_2$ was defined in (A.9). In order to separate one-loop and primitive two-loop subdivergences, we then decompose $\mathcal{F}_2^\Lambda$ into three subregions,

$$\begin{aligned}
\mathcal{F}_2^0 &= \mathcal{F}_2^\Lambda \cap \{\rho_2 \leq t + u_2^2\rho_2 \leq \Lambda_1\}, \\
\mathcal{F}_2^I &= \mathcal{F}_2^\Lambda \cap \{\rho_2 \leq \Lambda_1 \leq t + u_2^2\rho_2\}, \\
\mathcal{F}_2^{II} &= \mathcal{F}_2^\Lambda \cap \{\Lambda_1 \leq \rho_2 \leq t + u_2^2\rho_2\},
\end{aligned} \tag{B.70}$$

where $\Lambda_1 \ll \Lambda$ is a fiducial scale. One-loop subdivergences arise from integration over $\mathcal{F}_2^I$, while primitive divergence arises from integrating over $\mathcal{F}_2^{II}$. In extracting the divergences as $\Lambda \to \infty$, we can safely ignore terms proportional to powers of $\Lambda_1$, since they cancel in the sum over the three regions [107].

Let us first consider the divergences from region I. In this region, the variable $t$ is bounded by $\Lambda$ while $\rho$ is restricted to the fundamental domain $\mathcal{F}_{1,\Lambda_1}$. For the first $1 + N$ cosets of $\Gamma_{2,0}(N)\backslash Sp(4,\mathbb{Z})$ listed in (A.22), the charges $(Q_1, Q_2)$ whose contributions are not exponentially suppressed as $t \to \infty$ are those with $Q_2 = 0$. For those, the integral over $\sigma_1$ projects

$1/\Phi_{k-2}|_\gamma$ to its zero-mode $\psi_0|_\gamma$ in (A.59), while the remaining integral over $u_1, u_2$ projects the latter to its average value (A.73), with a factor of $1/2$ because of the element of $SL(2,\mathbb{Z})$ permuting them. The divergence from these $N+1$ cosets is then

$$-\frac{k}{32\pi}\int^{\Lambda}\frac{\mathrm{d}t}{t^3}\, t^{\frac{q}{2}-1}\mathrm{R.N.}\int_{\mathcal{F}_1}\frac{\mathrm{d}\rho_1\mathrm{d}\rho_2}{\rho_2^2}\sum_{\gamma_\rho\in SL(2,\mathbb{Z})/\Gamma_0(N)}\left[\frac{N^2 E_2(N\rho)-E_2(\rho)}{(N-1)\,\Delta_k(\rho)}\Gamma_{\Lambda_{p,q}}[P_{\langle ab,}]\delta_{cd\rangle}\right]\Bigg|_{\gamma_\rho}. \tag{B.71}$$

For the remaining $N^2 + N^3$ cosets, the representative $\gamma$ includes again the $N+1$ $\gamma_\rho$ elements again, times the $N$ transformations $\{S_\sigma, T_\sigma S_\sigma, \ldots, T_\sigma^{N-1}S_\sigma\}$, which requires a Poisson resummation over $Q_2$ before setting its dual to $0$, and the $N$ shifts $b$ in (A.22). The divergence is then of the same form as above, upon replacing $\psi_0$ by its image under $S_\sigma$, $N^{k/2}\hat{\psi}_0$ (A.59), and including a volume factor $|\Lambda_{p,q}^*/\Lambda_{p,q}|^{-\frac{1}{2}} = \upsilon N^{-\frac{k}{2}-2}$ from the Poisson resummation and a multiplicity factor $N^2$ from the transformations listed above:

$$-\frac{k\upsilon}{32\pi}\int^{\Lambda}\frac{\mathrm{d}t}{t^3}\, t^{\frac{q}{2}-1}\mathrm{R.N.}\int_{\mathcal{F}_1}\frac{\mathrm{d}\rho_1\mathrm{d}\rho_2}{\rho_2^2}\sum_{\gamma_\rho\in SL(2,\mathbb{Z})/\Gamma_0(N)}\left[\frac{E_2(\rho)-E_2(N\rho)}{(N-1)\,\Delta_k(\rho)}\Gamma_{\Lambda_{p,q}}[P_{\langle ab,}]\delta_{cd\rangle}\right]\Bigg|_{\gamma_\rho}. \tag{B.72}$$

For the perturbative $\nabla^2 F^4$ coupling in $D = 10 - q$ dimensions, the volume factor is $\upsilon = N$. After unfolding the integral to the domain $\mathcal{H}_1/\Gamma_0(N)$, the two contributions (B.71), (B.72) add up to

$$-\frac{k}{32\pi}\frac{\Lambda^{\frac{q-6}{2}}}{\frac{q-6}{2}}\mathrm{R.N.}\int_{\Gamma_0(N)\backslash\mathcal{H}_1}\frac{\mathrm{d}\rho_1\mathrm{d}\rho_2}{\rho_2^2}\frac{N\hat{E}_2(N\rho)+\hat{E}_2(\rho)}{\Delta_k(\rho)}\Gamma_{\Lambda_{p,q}}[P_{\langle ab,}]\delta_{cd\rangle} = -\frac{3}{8\pi}\frac{\Lambda^{\frac{q-6}{2}}}{\frac{q-6}{2}}\delta_{\langle ab,}F_{cd\rangle e}^{(p,q)\,e}, \tag{B.73}$$

where we recognized the coefficient of the divergence as the renormalized one-loop $F^4$ coupling by integrating by part, as in [22, §3.2], upon using the identity

$$D_{-k}\left(\frac{1}{\Delta_k(\rho)}\right) = \frac{k}{12}\frac{N\hat{E}_2(N\rho)+\hat{E}_2(\rho)}{\Delta_k(\rho)}, \tag{B.74}$$

where $D_w = \frac{i}{\pi}(\partial_\tau - \frac{iw}{2\tau_2})$ is the raising operator.

We now turn to the primitive two-loop divergence coming from the integral over $\mathcal{F}_2^{II}$. In this region, it is more convenient to use the variables $V, \tau$ defined in (A.9). The variable $V$ runs from $\tau_2/\Lambda$ to $1/\tau_2\Lambda_1$, while the variable $\tau$ takes values in the standard fundamental domain $\mathcal{F}_1/\mathbb{Z}_2$ of $GL(2,\mathbb{Z})$, truncated at $\tau_2 \leq \sqrt{\Lambda/\Lambda_1}$ [107]. The primitive divergence comes from the region $V \to 0$. For the first coset in (A.22), the contribution of all charge vectors with $Q_1 \neq 0$ or $Q_2 \neq 0$ are exponentially suppressed as $V \to 0$. For $(Q_1, Q_2) = (0, 0)$, the polynomial $P_{ab,cd}$ in (2.31) reduces to $3\delta_{\langle ab,}\delta_{cd\rangle}/(16\pi^2|\Omega_2|)$, and the integral over $\Omega_1$ projects $1/\Phi_{k-2}$ to its zero-mode $C_{k-2}(0,0,0) = \frac{48N}{N^2-1}$ in (A.49). For the second and third class of cosets in (A.22), the limit $V \to 0$ requires first performing a Poisson resummation over either $Q_1$ or $Q_2$, resulting in a volume factor of $|\Lambda_{p,q}^*/\Lambda_{p,q}|^{-\frac{1}{2}} = \upsilon N^{-\frac{k}{2}-2}$, and the integral over $\Omega_1$ projects $N^{\frac{k}{2}}/\tilde{\Phi}_{k-2}|_\gamma$ to its zero-mode $N^{k/2}\widetilde{C}_{k-2}(0,0,0) = -\frac{48N^{k/2}}{N^2-1}$ from (A.50), for each of the $N(N+1)$ cosets. Finally, for the fourth class of cosets in (A.22), the limit $V \to 0$ requires performing a Poisson resummation over both $Q_1$ and $Q_2$, resulting in a volume factor of $|\Lambda_{p,q}^*/\Lambda_{p,q}|^{-1} = \upsilon^2 N^{-k-4}$, and the integral over $\Omega_1$ projects $N^{k-2}/\Phi_{k-2}(\Omega/N)|_\gamma$ to its zero-mode after having used the identity (A.40), for each of the $N^3$ cosets. Adding up all contributions, we find

$$\frac{3\delta_{\langle ab,}\delta_{cd\rangle}}{16\pi^2}\mathrm{R.N.}\int_{\mathcal{F}_1/\mathbb{Z}_2}\frac{48\mathrm{d}\tau_1\mathrm{d}\tau_2}{(N^2-1)\tau_2^2}\int_{\tau_2/\Lambda}^{\tau_2/\Lambda_1}2V^2\mathrm{d}V\, V^{2-q}\left[N-(N+1)\frac{\upsilon}{N}+\frac{\upsilon^2}{N^2}\right]. \tag{B.75}$$

Setting $v = N$, the term in square bracket cancels, so the coefficient of the two-loop primitive divergence in fact vanishes.

Finally, it remains to consider a potential divergence from the separating degeneration. For generic values of $\rho, \sigma$ in $\mathcal{F}_2$, the integral around $v = 0$ is of the form $\int dv d\bar{v}/v^2$, which vanishes provided one integrates first over the angular direction in the $v$-plane. There can however be a divergence from the region $\rho_2, \sigma_2 \to \infty$ while $v \to 0$, where the genus-two curve degenerates into a figure-eight graph. For the first coset in (A.22), the contribution of all charge vectors with $Q_1 \neq 0$ or $Q_2 \neq 0$ are exponentially suppressed as $\rho_2, \sigma_2 \to \infty$. As shown in §A.6, the integral over $v_1$ gives rise to a delta-function $c_k(0)^2 \delta(v_2)$. To integrate this delta distribution it is convenient to unfold the integration domain of $\Omega_2$ near the cusp $|\Omega_2| \to \infty$, $\mathcal{P}_2/GL(2, \mathbb{Z})$ to $\mathcal{P}_2$, using the sum over $GL(2, \mathbb{Z})/\text{Dih}_4$ in (5.25), and taking into account the factor of 4 associated to $\text{Dih}_4$, the stabilizer of the singular locus $v = 0$. Equivalently one can think of the integral over $\mathcal{P}_2/GL(2, \mathbb{Z})$, and simply unfold the order four symmetry permuting $\sigma_2$ and $\rho_2$ and changing the sign of $v_2$. At $v_2 = 0$, $\sigma_2 = t$ and the integration domain is $\Lambda_1 \leq \rho_2 \leq \sigma_2 < \Lambda$, which after symmetrization gives the divergent contribution

$$-\frac{3k^2}{256\pi^3} \int^{\Lambda} \frac{d\rho_2}{\rho_2^3} \int^{\Lambda} \frac{d\sigma_2}{\sigma_2^3} (\rho_2 \sigma_2)^{\frac{q}{2}} \frac{\delta_{\langle ab} \delta_{cd\rangle}}{\rho_2 \sigma_2} \sim -\frac{3k^2}{256\pi^3} \left(\frac{\Lambda^{\frac{q-6}{2}}}{\frac{q-6}{2}}\right)^2 \delta_{\langle ab}, \delta_{cd\rangle} \, . \tag{B.76}$$

For the other cosets in (A.22), the zeroth Fourier-Jacobi coefficient behaves has $N^{\frac{k}{2}} \hat{\psi}_0(\rho, v)$ leading to $N^{\frac{k}{2}} c_k(0)^2 \delta(v_2)$, and $N^{k-2} \psi_0(\rho/N, v/N)$ leading to $N^{k-2} c_k(0)^2 \delta(v_2/N)$. The first contribution occurs from the trivial coset only; the second from $2N$ cosets because of the symmetry $\rho \leftrightarrow \sigma$, with an overall volume factor $vN^{-\frac{k}{2}-2}$; and the third from $N^3$ cosets corresponding to all shifts $(\frac{\rho+a}{N}, \frac{\sigma+b}{N}, \frac{v+c}{N})$, with an overall volume factor $v^2 N^{-k-4}$. Combining these terms and using $c_k(0) = k$, we find that the divergence from the figure-eight degeneration is

$$-\frac{3k^2}{256\pi^3} \frac{(N^2 + 2Nv + v^2)}{N^2} \left(\frac{\Lambda^{\frac{q-6}{2}}}{\frac{q-6}{2}}\right)^2 \delta_{\langle ab}, \delta_{cd\rangle} = -\frac{3k^2(1 + \frac{v}{N})^2}{256\pi^3} \left(\frac{\Lambda^{\frac{q-6}{2}}}{\frac{q-6}{2}}\right)^2 \delta_{\langle ab}, \delta_{cd\rangle} \, . \tag{B.77}$$

For $q = 6$, the divergent term $(\Lambda^{\frac{q-6}{2}}/\frac{q-6}{2})^2$ is replaced by $(\log \Lambda)^2$.

Combining these results, we can now define the renormalized integral (2.30) by subtracting all divergent contributions before taking the limit $\Lambda \to \infty$. In the case of the two-loop $\nabla^2 F^4$ couplings ($v = N$), we obtain

$$G_{ab,cd}^{(p,q)} = \lim_{\Lambda \to \infty} \Bigg[ \int_{\mathcal{F}_2^{\Lambda}} \frac{d^3\Omega_1 d^3\Omega_2}{|\Omega_2|^3} \sum_{\gamma \in \Gamma_{2,0}(N) \backslash Sp(4,\mathbb{Z})} \frac{\Gamma_{\Lambda_{p,q}}^{(2)} [P_{ab,cd}]}{\Phi_{k-2}(\Omega)} \Bigg|_{\gamma} + \frac{\Lambda^{\frac{q-6}{2}}}{\frac{q-6}{2}} \frac{3}{8\pi} \delta_{\langle ab}, F_{cd\rangle e}^{(p,q) e}$$
$$+ \left(\frac{\Lambda^{\frac{q-6}{2}}}{\frac{q-6}{2}}\right)^2 \frac{3k^2}{64\pi^3} \delta_{\langle ab}, \delta_{cd\rangle} \Bigg] \, . \tag{B.78}$$

For $q = 6$, the $\mathcal{O}(\Lambda^{\frac{q-6}{2}})$ and $\mathcal{O}(\Lambda^{q-6})$ divergences become logarithmic and doubly logarithmic,

$$\widehat{G}_{ab,cd}^{(2k-2,6)} = \lim_{\Lambda \to \infty} \Bigg[ \int_{\mathcal{F}_2^{\Lambda}} \frac{d^3\Omega_1 d^3\Omega_2}{|\Omega_2|^3} \sum_{\gamma \in \Gamma_{2,0}(N) \backslash Sp(4,\mathbb{Z})} \frac{\Gamma_{\Lambda_{2k-2, 6}}^{(2)} [P_{ab,cd}]}{\Phi_{k-2}(\Omega)} \Bigg|_{\gamma} + \log \Lambda \frac{3}{8\pi} \delta_{\langle ab}, \widehat{F}_{cd\rangle e}^{(2k-2,6) e}$$
$$+ (\log \Lambda)^2 \frac{3k^2}{64\pi^3} \delta_{\langle ab}, \delta_{cd\rangle} \Bigg] \, , \tag{B.79}$$

where $F^{(p,q)}_{abcd}$ is the regularized integral (B.13).

The renormalization of the couplings $F_{abcd}$ and $G_{ab,cd}$ is in fact consistent with supergravity computations [83], as we now explain. Recall that the complete string theory amplitude can be obtained by performing a functional integral over the fields of $\mathcal{N}=4$ supergravity with $2k-2$ vector multiplets, weighted by the Wilsonian effective action computed in string theory. This Wilsonian action can be defined by imposing an infrared cutoff $\Lambda$ on the moduli space of complex structures, identified with the ultra-violet cutoff in supergravity. It follows that the $\Lambda$-dependent couplings

$$
\begin{aligned}
F^{(2k-2,6)}_{abcd}(\Lambda) &= F^{(2k-2,6)}_{abcd} + \frac{3k}{8\pi^2}\log\Lambda\,\delta_{(ab}\delta_{cd)}\,, \\
G^{(2k-2,6)}_{ab,cd}(\Lambda) &= G^{(2k-2,6)}_{ab,cd} - \log\Lambda\,\frac{3}{8\pi}\delta_{\langle ab,}F^{(2k-2,6)e}_{cd\rangle e} - (\log\Lambda)^2\frac{3k^2}{64\pi^3}\delta_{\langle ab,}\delta_{cd\rangle}\,,
\end{aligned}
\tag{B.80}
$$

define a bare Lagrangian

$$
\begin{aligned}
\mathcal{L}(\Lambda) = \frac{2}{\kappa^2}\mathcal{R} &- \frac{1}{4}\delta_{ab}F^a F^b + \tfrac{1}{8}(\tfrac{\kappa}{2})^4 F^{(2k-2,6)}_{abcd}(\Lambda)t_8 F^a F^b F^c F^d \\
&+ \tfrac{1}{8\pi}(\tfrac{\kappa}{2})^6 G^{(2k-2,6)}_{ab,cd}(\Lambda)\,t_8\nabla F^a\nabla F^b F^c F^d + \dots
\end{aligned}
\tag{B.81}
$$

such that the UV divergences in the path integral cancel at this order. These divergences cancel for any functions $F^{(2k-2,6)}_{abcd}$ and $G^{(2k-2,6)}_{ab,cd}$ satisfying their respective differential constraints. Upon setting $F^{(2k-2,6)}_{abcd}$ and $G^{(2k-2,6)}_{ab,cd}$ to zero in (B.80), one reproduces precisely the counter-terms computed in [83] in four dimensions. The variation of $\mathcal{L}(\Lambda)$ with respect to $F^{(2k-2,6)}_{abcd}$ is interpreted in supergravity as the form factor for the operator $t_8 F^4$ (at zero momentum and properly supersymmetrized). Similarly, the variation of $\mathcal{L}(\Lambda)$ with respect to $G^{(2k-2,6)}_{ab,cd}$ is the form factor for the operator $t_8\nabla^2 F^4$. Because (3.20) does not admit a constant homogeneous solution for $q=6$, there cannot be any genuine 2-loop divergence proportional to $\delta_{\langle ab,}\delta_{cd\rangle}$ in $\mathcal{N}=4$ supergravity. The 2-loop divergence proportional to $(\log\Lambda)^2$ in (B.80) is therefore a consequence of the 1-loop divergence, via the renormalization group equation

$$
\Lambda\frac{\mathrm{d}}{\mathrm{d}\Lambda}G^{(2k-2,6)}_{ab,cd}(\Lambda) = -\frac{3}{4\pi}\delta_{\langle ab,}F^{(2k-2,6)\,e}_{cd\rangle e}(\Lambda)\,.
\tag{B.82}
$$

This is consistent with the supergravity analysis in [83, §5.A], where the two-loop divergence originates entirely from figure-eight supergravity diagrams (shown in Figure 1ii), for which the subdivergence is proportional to the 1-loop counter-term form factor.

Let us now briefly discuss the regularization of the integral (B.69) in the case where the lattice $\Lambda_{p,q}$ is the non-perturbative Narain lattice (2.3). In this case, the volume factor $\upsilon$ is equal to 1. In this case, the cancellation in (B.75) still takes place in the maximal rank case since the zero-th Fourier coefficient of $1/\Phi_{10}$ vanishes from (A.48), but it no longer holds for CHL models with $N=2,3,5,7$. Setting $\upsilon=1$ in the previous computations, we now get

$$
\begin{aligned}
G^{(p,q)}_{ab,cd} = \lim_{\Lambda\to\infty}\Big[&\int_{\mathcal{F}^\Lambda_2}\frac{\mathrm{d}^3\Omega_1\mathrm{d}^3\Omega_2}{|\Omega_2|^3}\sum_{\gamma\in\Gamma_{2,0}(N)\backslash Sp(4,\mathbb{Z})}\frac{\Gamma^{(2)}_{\Lambda_{p,q}}[P_{ab,cd}]}{\Phi_{k-2}(\Omega)}\Big|_\gamma + \frac{27}{\pi^2 N^2}\frac{\Lambda^{q-6}}{(q-6)^2}\delta_{\langle ab,}\delta_{cd\rangle} \\
&- \frac{9(N-1)}{\pi^2 N^2}\frac{\Lambda^{q-5}}{q-5}\delta_{\langle ab,}\delta_{cd\rangle}\,\mathrm{R.N.}\int_{\mathcal{F}_1}\frac{\mathrm{d}\tau_1\mathrm{d}\tau_2}{\tau_2^2}\tau_2^{5-q} + \frac{3}{2\pi N}\frac{\Lambda^{\frac{q-6}{2}}}{\frac{q-6}{2}}\,{}^\varsigma G^{(p,q)}_{\langle ab,}\delta_{cd\rangle})\Big]\,,
\end{aligned}
\tag{B.83}
$$

where ${}^\varsigma G^{(p,q)}_{ab}$ denotes the regularized integral (B.17). The maximal rank case is obtained by setting $N=1$, and ${}^\varsigma G^{(p,q)}_{ab} = G^{(p,q)}_{ab}$. Of course, the case relevant for the non-perturbative $\nabla^2(\nabla\phi)^4$ coupling in $D=3$ corresponds to $q=8$, in which case there are power-like divergences but no logarithmic divergence.

### B.2.5 Anomalous terms in the differential equation for $G_{ab,cd}$

In section 3.3 we established that the renormalized integral $G_{ab,cd}^{(p,q)}$ satisfies the differential equation (3.20), with a quadratic source term originating from the separating degeneration locus $v = 0$. In this section we take into account the boundary of the regularized domain $\mathcal{F}_2^\Lambda$ and show that the equation indeed holds for the renormalized couplings at generic values of $q$. For $q = 5$ with $v \neq N$ and $q = 6$ we find additional linear source terms from the non-separating degeneration. For the perturbative amplitude in four dimensions, $q = 6$, $v = N$, these linear term originate from the mixing between the analytic and the non-analytic components of the amplitude. Our analysis parallels that of the $D^6 \mathcal{R}^4$ couplings in [107, §3.3].

From the $t = \Lambda$ boundary of the region $\mathcal{F}_2^I$ defined in (B.70), the leading contribution of the polynomial insertion is given by

$$\Omega_2^{2r}\Omega_2^{2s}e^{-\frac{\Delta_2}{8\pi}}Q_{Ler}Q_{Lfs}e^{\frac{\Delta_2}{8\pi}}P_{ab,cd}\Big|_{Q_2=0} = \frac{3}{16\pi^2}\left(\delta_{ef}\delta_{\langle ab,}P_{cd\rangle} + 2\delta_{e\langle b}\delta_{|f|b,}P_{cd\rangle}\right)+\mathcal{O}(t^{-1}), \quad \text{(B.84)}$$

so using (A.73), with a factor $1/2$ due to the $\mathbb{Z}_2$ symmetry $(u_1, u_2) \to (-u_1, -u_2)$ at the cusp, we find that the right-hand side of (3.62) receives an additional contribution given by

$$-\frac{k\Lambda^{\frac{q-6}{2}}}{64\pi}\text{R.N.}\int_{\mathcal{F}_1}\frac{d\rho_1 d\rho_2}{\rho_2^2}\sum_{\gamma_\rho\in SL(2,\mathbb{Z})/\Gamma_0(N)}\left[\frac{N^2\hat{E}_2(N\rho)-\hat{E}_2(\rho)}{(N-1)\,\Delta_k(\rho)}\Gamma_{\Lambda,p,q}[P_{\langle ab,}](\delta_{cd\rangle}\delta_{ef} + 2\delta_{c|e|}\delta_{d\rangle f})\right]\Big|_{\gamma_\rho}$$

$$-\frac{kv\Lambda^{\frac{q-6}{2}}}{64\pi}\text{R.N.}\int_{\mathcal{F}_1}\frac{d\rho_1 d\rho_2}{\rho_2^2}\sum_{\gamma_\rho\in SL(2,\mathbb{Z})/\Gamma_0(N)}\left[\frac{\hat{E}_2(\rho)-\hat{E}_2(N\rho)}{(N-1)\,\Delta_k(\rho)}\Gamma_{\Lambda,p,q}[P_{\langle ab,}](\delta_{cd\rangle}\delta_{ef} + 2\delta_{c|e|}\delta_{d\rangle f})\right]\Big|_{\gamma_\rho}, \quad \text{(B.85)}$$

where the first and second line results respectively from cosets elements $(\gamma, 1)$ and $(\gamma, S_\sigma) \in (SL(2, \mathbb{Z})/\Gamma_0(N))_\rho \times (SL(2\mathbb{Z})/\Gamma_0(N))_\sigma$, while other terms in the coset sum are annihilated by integration over $\sigma_1, v_1 \in [-\frac{1}{2}, \frac{1}{2}]$. The sum (B.85) can be rewritten in terms of the regularized integral $G_{ab}^{(p,q)}$ as

$$-\frac{k(v-1)\Lambda^{\frac{q-6}{2}}}{64\pi(N-1)}\left(\delta_{ef}\delta_{\langle ab}G_{cd\rangle}^{(p,q)} + 2\delta_{e\langle a}\delta_{b,|f|}G_{cd\rangle}^{(p,q)}\right)$$

$$-\frac{k(N^2-v)\Lambda^{\frac{q-6}{2}}}{64\pi N(N-1)}\left(\delta_{ef}\delta_{\langle ab}{}^\varsigma G_{cd\rangle}^{(p,q)} + 2\delta_{e\langle a}\delta_{b,|f|}{}^\varsigma G_{cd\rangle}^{(p,q)}\right). \quad \text{(B.86)}$$

This terms gives a finite correction to the differential equation for $q = 6$.

The right-hand side of (3.62) also receives contributions from the boundary of region $\mathcal{F}_2^{II}$ in (B.70), where the leading contribution of the polynomial insertion is

$$(\Omega_2)_{rs}e^{-\frac{\Delta_2}{8\pi}}Q_{Le}^r Q_{Lf}^s e^{\frac{\Delta_2}{8\pi}}P_{ab,cd}\Big|_{Q_1=Q_2=0} = -\frac{3}{32\pi^3|\Omega_2|}\left(\delta_{ef}\delta_{\langle ab,}\delta_{cd\rangle} + 2\delta_{e\langle a}\delta_{|f|b,}\delta_{cd\rangle}\right)+\mathcal{O}(\Omega_2^{-1}). \quad \text{(B.87)}$$

Its contribution to the right hand side of (3.62) thus reduces to [31]

$$-\frac{3}{32\pi^2}\text{R.N.}\int_{\mathcal{F}_1/\mathbb{Z}_2}\frac{d\tau_1 d\tau_2}{\tau_2^2}\int_{\frac{\tau_2}{\Lambda}}2dV\frac{\partial}{\partial V}\frac{1}{V^3}\left(\frac{|\Omega_2|^{\frac{q}{2}}}{|\Omega_2|^3}\frac{\delta_{ef}\delta_{\langle ab}\delta_{cd\rangle} + 2\delta_{e\langle a}\delta_{|f|b,}\delta_{cd\rangle}}{|\Omega_2|}\right)$$

$$\times\left(\frac{2k}{N-1}\left[N-\frac{v}{N}(N+1)+\frac{v^2}{N^2}\right]-\frac{1}{4\pi}k^2\delta\left(\frac{\tau_1}{V\tau_2}\right)\left[1+2\frac{v}{N}+\frac{v^2}{N^2}\right]\right). \quad \text{(B.88)}$$

---

[31]Where one uses $2id^3\Omega_2\frac{\partial}{\partial\hat{\Omega}_{rs}}\left((\Omega_2)_{rt}(\Omega_2)_{su}(\Omega_2^{-1})^{tu}X(\Omega_2)\right) = \frac{2dVd\tau_1 d\tau_2}{\tau_2^2}\frac{\partial}{\partial V}\frac{X(\Omega_2)}{V^3}$ at the boundary $V = \frac{\tau_2}{\Lambda}$.

In (B.88) we kept the constant term in the Fourier expansions of $1/\Phi_{k-2}$ and we used $\partial/\partial\bar{\Omega} \sim -\frac{i}{4}V\Omega_2^{-1}\partial/\partial V$. On the boundary at $V = \tau_2/\Lambda$, the first term in (B.88) gives

$$\Lambda^{q-5}\frac{3k(N-\upsilon)(1-\frac{\upsilon}{N^2})}{8\pi^2(N-1)}(\delta_{ef}\delta_{\langle ab}\delta_{cd\rangle} + 2\delta_{e\langle a}\delta_{b,|f|}\delta_{cd\rangle})\,\text{R.N.}\int_{\mathcal{F}_1/\mathbb{Z}_2}\frac{d\tau_1 d\tau_2}{\tau_2^2}\tau_2^{5-q}, \quad \text{(B.89)}$$

which vanishes in the perturbative case, $\upsilon = N$. The second term in (B.88) integrates to

$$-\frac{\Lambda^{q-6}}{q-6}\frac{3k^2(\delta_{ef}\delta_{\langle ab}\delta_{cd\rangle} + 2\delta_{e\langle a}\delta_{b,|f|}\delta_{cd\rangle})}{128\pi^3}\Big(1 + \frac{\upsilon}{N}\Big)^2. \quad \text{(B.90)}$$

The case $q = 6$ must be computed separately and turns out to give zero. Finally, the quadratic term in the second line of (3.66) can be written using the regularized genus-one integral $F^{(p,q)}_{abcd}$ (B.15) as

$$-\frac{3\pi}{2}F^{(p,q)}_{|e)k\langle ab,}(\Lambda)F^{(p,q)k}_{cd\rangle}{}_{(f|}(\Lambda) = -\frac{3\pi}{2}F^{(p,q)}_{|e)k\langle ab,}F^{(p,q)k}_{cd\rangle}{}_{(f|} - \frac{3k(1+\frac{\upsilon}{N})}{16\pi}\frac{\Lambda^{\frac{q-6}{2}}}{\frac{q-6}{2}}\delta_{\langle ab,}F^{(p,q)}_{cd\rangle ef} \quad \text{(B.91)}$$

$$-\frac{3k^2(1+\frac{\upsilon}{N})^2}{512\pi^3}\Big(\frac{\Lambda^{\frac{q-6}{2}}}{\frac{q-6}{2}}\Big)^2\Big(\delta_{ef}\delta_{\langle ab,}\delta_{cd\rangle} + 2\delta_{e\langle a}\delta_{b,|f|}\delta_{cd\rangle}\Big).$$

Using the action of the operator (3.59) on the tensor defining the counter-terms of $G^{(p,q)}_{ab,cd}$,

$$\Delta_{ef}\delta_{\langle ab}G^{(p,q)}_{cd\rangle} = \frac{q-6}{4}\Big(\delta_{ef}\delta_{\langle ab,}G^{(p,q)}_{cd\rangle} + 2\delta_{e\langle(a}\delta_{b),|f|}G^{(p,q)}_{cd\rangle}\Big) + 6\delta_{\langle ab,}F^{(p,q)}_{cd\rangle ef}, \quad \text{(B.92)}$$

$$\Delta_{ef}\delta_{\langle ab}{}^{\varsigma}G^{(p,q)}_{cd\rangle} = \frac{q-6}{4}\Big(\delta_{ef}\delta_{\langle ab,}{}^{\varsigma}G^{(p,q)}_{cd\rangle} + 2\delta_{e\langle(a}\delta_{b),|f|}{}^{\varsigma}G^{(p,q)}_{cd\rangle}\Big) + 6\delta_{\langle ab,}F^{(p,q)}_{cd\rangle ef}, \quad \text{(B.93)}$$

$$\Delta_{ef}\delta_{\langle ab,}\delta_{cd\rangle} = \frac{q-5}{2}\Big(\delta_{ef}\delta_{\langle ab,}\delta_{cd\rangle} + 2\delta_{e\langle(a}\delta_{b),|f|}\delta_{cd\rangle}\Big), \quad \text{(B.94)}$$

one finds that all $\Lambda$ dependent terms cancel in the differential equation for the renormalized coupling, such that for generic $q$,

$$\Delta_{ef}\Big(G^{(p,q)}_{ab,cd}(\Lambda) + \frac{k}{32\pi}\frac{\Lambda^{\frac{q-6}{2}}}{\frac{q-6}{2}}\delta_{\langle ab}\Big(\frac{\upsilon-1}{N-1}G^{(p,q)}_{cd\rangle} + \frac{N-\frac{\upsilon}{N}}{N-1}{}^{\varsigma}G^{(p,q)}_{cd\rangle}\Big) + \frac{3k^2(1+\frac{\upsilon}{N})^2}{256\pi^3}\Big(\frac{\Lambda^{\frac{q-6}{2}}}{\frac{q-6}{2}}\Big)^2\delta_{\langle ab,}\delta_{cd\rangle}$$

$$-\frac{3k}{4\pi}\frac{\Lambda^{q-5}}{q-5}\frac{(N-\upsilon)(1-\frac{\upsilon}{N^2})}{N-1}\delta_{\langle ab,}\delta_{cd\rangle}\text{R.N.}\int_{\mathcal{F}_1/\mathbb{Z}_2}\frac{d\tau_1 d\tau_2}{\tau_2^2}\tau_2^{5-q}\Big)$$

$$= -\frac{3\pi}{2}F^{(p,q)}_{|e)k\langle ab,}F^{(p,q)k}_{cd\rangle}{}_{(f|}. \quad \text{(B.95)}$$

The cases featuring logs must be treated separately. Here we shall only discuss the case of the perturbative lattice in four dimensions, i.e. $\upsilon = N$ and $q = 6$, which is physically relevant.

Because the first term proportional to $q-6$ in (B.92) vanishes at $q = 6$, it does not cancel the finite contribution from (B.86) and one gets an additional linear source term in the equation. The computation of the anomalous terms from the counter-term in $G^{(2k-2,6)}_{ab} + {}^{\varsigma}G^{(2k-2,6)}_{ab}$ involves the detailed analysis of the integration by part in the boundary between regions $\mathcal{F}_2^I$ and $\mathcal{F}_2^{II}$. Since this boundary is artificial, these anomalous terms must cancel other contributions from (B.85) and (B.88), such that one can assume that $G^{(2k-2,6)}_{ab} + {}^{\varsigma}G^{(2k-2,6)}_{ab}$ satisfies the naive differential equation (B.92), ignoring the anomalous source term in (B.18). This prescription is in fact necessary for the differential equation to be well defined on the renormalized couplings. In this way we obtain

$$\Delta_{ef}\widehat{G}^{(2k-2,6)}_{ab,cd} = -\frac{3\pi}{2}\widehat{F}^{(2k-2,6)}_{|e)k\langle ab,}\widehat{F}^{(2k-2,6)k}_{cd\rangle}{}_{(f|} - \frac{3}{16\pi}\Big(\delta_{ef}\delta_{\langle ab,}\widehat{F}^{(2k-2,6)k}_{cd\rangle k} + 2\delta_{e\langle(a}\delta_{b),|f|}\widehat{F}^{(2k-2,6)k}_{cd\rangle k}\Big), \quad \text{(B.96)}$$

where we recall that $\Delta_{ef}$ is a shorthand for the operator in (3.59).

### B.3 Loci of enhanced gauge symmetry

Even after regulating infrared divergences occurring at generic points on $G_{p,q}$, further divergences may occur on loci of enhanced gauge symmetry, where perturbative 1/2-BPS states become massless. Divergences from region $\mathcal{F}_2^I$ in (B.70) occur from contributions of lattice vectors $Q_2 \in \Lambda$ such that $Q_2^2 = 2$. For such vectors, the integral over $\sigma_1 \in [0,1]$ picks up the polar term in the Fourier-Jacobi expansion (A.57) of $1/\Phi_{k-2}$, contributing a term of the form

$$
\int^\infty dt \, t^{\frac{q}{2}-3} e^{-2\pi t Q_{2R}^2} \times \int_{\mathcal{F}_1} \frac{d\rho_1 d\rho_2}{\rho_2^2} \int_{[0,1]^2} du_1 \, du_2 \, |\rho_2|^{q/2}
$$
$$
\times \sum_{Q_1 \in \Lambda_{p,q}} P_{ab,cd} \, q^{\frac{1}{2}Q_{1L}^2} \bar{q}^{\frac{1}{2}Q_{1R}^2} e^{-\pi \rho_2^2 Q_{2R}^2 + 2\pi i (v Q_{1L} \cdot Q_{2L} - \bar{v} Q_{1R} \cdot Q_{2R})} \frac{\eta^6}{\Delta_k(\rho) \, \theta_1^2(\rho, v)}
\tag{B.97}
$$

to the modular integral $G_{ab,cd}^{(p,q)}$. The integral over $t$ diverges on the codimension $q$ locus where $|Q_{2R}| \to 0$, corresponding to 1/2-BPS states with charge $\pm Q_2$ becoming massless. This is a familiar phenomenon in perturbative heterotic string theory, where such BPS states can be viewed as W-bosons for a $SU(2)$ gauge symmetry which spontaneously broken away from the locus where $|Q_{2R}| = 0$. Near the singular locus, the genus-two integral diverges as a sum of powers of the mass $\mathcal{M} = \sqrt{2}|Q_{2R}|$, weighted by the genus-one modular integral appearing in (B.97), which can interpreted as the four-point amplitude with two massless and two massive gauge bosons. Note that this genus-one integral does not suffer from any divergence from the lattice vector $Q_1 = Q_2$, since the polynomial $P_{ab,cd}$ in representation ⊞ vanishes when $Q_1$ and $Q_2$ are collinear. Of course, similar gauge symmetry enhancements arise from vectors $Q_2 \in \Lambda_{p,q}$ with $Q_2^2 = 2/N$, due to the polar term in the Fourier-Jacobi expansion of the images of $1/\Phi_{k-2}$ under $\Gamma_{2,0}(N) \backslash Sp(4, \mathbb{Z})$.

In addition, the modular integral $G_{ab,cd}^{(p,q)}$ has further singularities from region $\mathcal{F}_2^{II}$, due to polar terms of the form $q_1^{-N_1} q_2^{-N_2} q_3^{-N_3}$ in the Fourier expansion (A.49) of $1/\Phi_{k-2}$, with $N_1, N_2, N_3 < 0$. The integral over $\Omega_1$ picks up contributions of pairs of vectors $(Q_1, Q_2) \in \Lambda_{p,q} \oplus \Lambda_{p,q}$ satisfying the level-matching conditions

$$
Q_1^2 - 2N_1 = Q_2^2 - 2N_2 = Q_3^2 - 2N_3 = 0,
\tag{B.98}
$$

where we denote $Q_3 = Q_1 + Q_2$. The remaining integral over $\Omega_2$ is of then the form

$$
\int \frac{dL_1 dL_2 dL_3}{(L_1 L_2 + L_2 L_3 + L_3 L_1)^{\frac{6-q}{2}}} P_{ab,cd} \, e^{-2\pi \left( L_1 Q_{1R}^2 + L_2 Q_{2R}^2 + L_3 Q_{3R}^2 \right)},
\tag{B.99}
$$

which for $q = 6$ has a leading singularity in

$$
\int dL_1 dL_2 dL_3 P_{ab,cd} \, e^{-2\pi \left( L_1 Q_{1R}^2 + L_2 Q_{2R}^2 + L_3 Q_{3R}^2 \right)} \sim \frac{\varepsilon_{rt} \varepsilon_{su} Q_{L(a}^r Q_{Lb)}^s Q_{L(c}^t Q_{Ld)}^u}{8\pi^3 Q_{1R}^2 Q_{2R}^2 Q_{3R}^2}.
\tag{B.100}
$$

This integral is singular on the codimension $q$ locus where $Q_{iR}^2 = 0$ for one index $i \in \{1, 2, 3\}$, but the corresponding divergence is covered by region I. Genuine new divergences occur in codimension $2q$ where $Q_{1R}^2 = Q_{2R}^2 = 0$ for two distinct indices, in which case $Q_{3R}^2$ automatically vanishes. The latter occurs for $(N_1, N_2, N_3) = (1, 1, 1)$ and corresponds to a $SU(3)$ gauge symmetry enhancement. Of course, similar divergences arise from pairs of vectors $(Q_1, Q_2) \in \Lambda_{p,q}^* \oplus \Lambda_{p,q}^*$ due to the polar terms in the Fourier expansion of the images of $1/\Phi_{k-2}$ under $\Gamma_{2,0}(N) \backslash Sp(4, \mathbb{Z})$. It would be interesting to recover (B.100) from a two-loop computation in a super-Yang-Mills theory with $SU(3)$ gauge group.

# C   Composite 1/4-BPS states, and instanton measure

In this Appendix our main aim is to prove Eqs (5.85) and (5.92), which play a central role in our analysis of the decompactification limit in §5. In particular, they ensure the consistency of the 1/4-BPS Abelian Fourier coefficients of $G_{ab,cd}$ with the differential equation (2.26), (3.20), and the consistency of the helicity supertrace (2.14) with wall-crossing, generalizing the consistency checks of [30] to arbitrary charges $\Gamma$. Specifically, we show that the summation measure $\bar{c}(Q, P; \Omega_2)$ for 1/4-BPS Abelian Fourier coefficients of $G_{ab,cd}$ decomposes into an $\Omega_2$-independent part associated to single-centered 1/4-BPS black holes, and a sum over all possible splittings of a 1/4-BPS charge vector $\Gamma = \Gamma_1 + \Gamma_2$ into 1/2-BPS charges, $\Gamma_1$ and $\Gamma_2$, weighted by the product $\bar{c}(\Gamma_1)\bar{c}(\Gamma_2)$ of the summation measures for 1/2-BPS black holes.

We start by describing the possible splittings of a 1/4-BPS charge $\Gamma = (Q, P)$ into 1/2-BPS constituents. Assuming an Ansatz of the form $\Gamma_1 = (p', r')(sQ - qP + tR)$ and $\Gamma_2 = (q', s')(pP - rQ + uR)$ for rational coefficients and linearly independent charges $(Q, P, R)$, with $R$ an arbitrary auxiliary charge, it is easy to find that the condition $\Gamma = \Gamma_1 + \Gamma_2$ fixes $t = u = 0$ and $p', r' q', s'$ such that

$$\binom{Q_1}{P_1} = \binom{p}{r}\frac{sQ - qP}{ps - qr}, \quad \binom{Q_2}{P_2} = \binom{q}{s}\frac{pP - rQ}{ps - qr}. \tag{C.1}$$

This splitting is conveniently parametrized by the a non-degenerate matrix $B = \left(\begin{smallmatrix} p & q \\ r & s \end{smallmatrix}\right) \in M_2(\mathbb{Z})$, such that

$$\binom{Q_1}{P_1} = B\pi_1 B^{-1}\binom{Q}{P}, \quad \binom{Q_2}{P_2} = B\pi_2 B^{-1}\binom{Q}{P}, \tag{C.2}$$

where $\pi_1 = \left(\begin{smallmatrix} 1 & 0 \\ 0 & 0 \end{smallmatrix}\right)$ and $\pi_2 = \left(\begin{smallmatrix} 0 & 0 \\ 0 & 1 \end{smallmatrix}\right)$. To parametrize the possible splittings bijectively one must factorize out the stabilizer $\text{Stab}(\pi_i)$ of $\pi_1$ and $\pi_2$ in $M_2(\mathbb{Z})$ up to permutation, *i.e.*

$$\text{Stab}(\pi_i) = \left\{ \left(\begin{smallmatrix} d_1 & 0 \\ 0 & d_2 \end{smallmatrix}\right), \left(\begin{smallmatrix} 0 & 1 \\ 1 & 0 \end{smallmatrix}\right) \right\}. \tag{C.3}$$

All splittings of a charge $\Gamma$ are therefore classified by the set of matrices $B \in M_2(\mathbb{Z})/\text{Stab}(\pi_i)$. Decomposing the matrix $B$ as

$$\begin{pmatrix} p & q \\ r & s \end{pmatrix} = \gamma \cdot \begin{pmatrix} p' & j \\ 0 & k \end{pmatrix}, \qquad\qquad \gamma \in GL(2, \mathbb{Z}), \quad p' > 0 \quad 0 \le j < k,$$
$$= \gamma \cdot \begin{pmatrix} 1 & \frac{j}{\gcd(j,k)} \\ 0 & \frac{k}{\gcd(j,k)} \end{pmatrix}\begin{pmatrix} p' & 0 \\ 0 & \gcd(j,k) \end{pmatrix}, \tag{C.4}$$

and using $\text{Stab}(\pi_i) \cap GL(2,\mathbb{Z}) = \text{Dih}_4$ one can always choose $\gamma \in GL(2,\mathbb{Z})/\text{Dih}_4$.[32] We conclude that the possible splittings are in one-to-one correspondence with the elements of

$$M_2(\mathbb{Z})/\text{Stab}(\pi_i) = \left\{ \gamma \cdot \begin{pmatrix} 1 & j' \\ 0 & k' \end{pmatrix}, \qquad \gamma \in GL(2,\mathbb{Z})/\text{Dih}_4, \quad 0 \le j' < k', \quad (j', k') = 1 \right\}, \tag{C.5}$$

such that the quantization condition $B\pi_i B^{-1}\Gamma \in \Lambda_m^* \oplus \Lambda_m$, $i = 1, 2$ on the charges of the two constituents is obeyed. It suffices to check this condition for $i = 1$, since the sum of the two is by assumption in $\Lambda_m^* \oplus \Lambda_m$.

---

[32]One checks indeed that the quotient by $\text{Dih}_4$ passes to the right of $\gamma$, by changing the representatives $\gamma$ and $j/\gcd(j,k)$ for $\left(\begin{smallmatrix} 0 & 1 \\ 1 & 0 \end{smallmatrix}\right) \in \text{Dih}_4$.

## C.1 Maximal rank

In the maximal rank case the condition $B\pi_1 B^{-1}\Gamma \in \Lambda_m^* \oplus \Lambda_m$ reduces to $\frac{aP-cQ}{k'} \in \Lambda_m$, with

$$(Q_1, P_1) = (a, c)\left(dQ - bP - \frac{j'}{k'}(aP - cQ)\right), \qquad (Q_2, P_2) = \left(\frac{j'}{k'}(a, c) + (b, d)\right)(aP - cQ). \quad \text{(C.6)}$$

These splittings are all related by $GL(2, \mathbb{Z})$ to a canonical splitting

$$\binom{Q}{P} = \binom{1}{0}\left(Q - \frac{j'}{k'}P\right) + \binom{j'}{k'}\frac{1}{k'}P, \qquad P/k' \in \Lambda_m. \quad \text{(C.7)}$$

Denoting by

$$\Delta\bar{C}(Q, P; \Omega_2) = \bar{C}(Q, P; \Omega_2) - \sum_{\substack{A \in M_2(\mathbb{Z})/GL(2,\mathbb{Z}) \\ A^{-1}\Gamma \in \Lambda_{22,6} \oplus \Lambda_{22,6}}} |A| \, C^F\left[A^{-1}\left(\begin{smallmatrix} -Q^2 & -Q\cdot P \\ -Q\cdot P & -P^2 \end{smallmatrix}\right)A^{-\mathsf{T}}\right] \quad \text{(C.8)}$$

the contribution from the poles of $1/\Phi_{10}$ on the second line of (5.25) to the measure factor (5.74) we thus find

$$\Delta\bar{C}(Q, P; \Omega_2) = \sum_{\substack{A \in M_2(\mathbb{Z})/\mathrm{Dih}_4 \\ A^{-1}\Gamma \in \Lambda_{22,6} \oplus \Lambda_{22,6}}} |A| \, c\left(-\frac{([A^{-1}\Gamma]_1)^2}{2}\right) c\left(-\frac{([A^{-1}\Gamma]_2)^2}{2}\right) \quad \text{(C.9)}$$

$$\times\left(-\frac{\delta([A^{\mathsf{T}}\Omega_2 A]_{12})}{4\pi} + \frac{[A^{-1}\Gamma]_1 \cdot [A^{-1}\Gamma]_2}{2}\left(\mathrm{sign}([A^{-1}\Gamma]_1 \cdot [A^{-1}\Gamma]_2) - \mathrm{sign}([A^{\mathsf{T}}\Omega_2 A]_{12})\right)\right),$$

where we combined the sum over $A \in M_2(\mathbb{Z})/GL(2, \mathbb{Z})$ and the sum over $\gamma \in GL(2, \mathbb{Z})/\mathrm{Dih}_4$ into the sum over $A\gamma \in M_2(\mathbb{Z})/\mathrm{Dih}_4$ that we call $A$ again, Further decomposing the sum over $A$ as

$$A = \gamma \cdot \begin{pmatrix} 1 & \frac{j'}{k'} \\ 0 & 1 \end{pmatrix}\begin{pmatrix} d_1 & 0 \\ 0 & d_2 \end{pmatrix} = \hat{B}\begin{pmatrix} d_1 & 0 \\ 0 & d_2 \end{pmatrix}, \quad \text{(C.10)}$$

with $k'|d_2$, and $\hat{B} = B\begin{pmatrix} 1 & 0 \\ 0 & |B|^{-1} \end{pmatrix}$ parametrizing the splittings, one obtains

$$\Delta\bar{C}(Q, P; \Omega_2) = \sum_{\substack{B \in M_2(\mathbb{Z})/\mathrm{Dih}_4 \\ \hat{B}^{-1}\Gamma \in \Lambda_m \oplus \Lambda_m}} \sum_{\substack{d_1 \geq 1 \\ \Gamma_1/d_1 \in \Lambda_m \oplus \Lambda_m}} c\left(-\frac{\gcd(Q_1^2, P_1^2, Q_1 \cdot P_1)}{2d_1^2}\right) \sum_{\substack{d_2 \geq 1 \\ \Gamma_2/d_2 \in \Lambda_m \oplus \Lambda_m}} c\left(-\frac{\gcd(Q_2^2, P_2^2, Q_2 \cdot P_2)}{2d_2^2}\right)$$

$$\times\left(-\frac{\delta([\hat{B}^{\mathsf{T}}\Omega_2\hat{B}]_{12})}{4\pi} + \frac{\langle\Gamma_1, \Gamma_2\rangle}{2}\left(\mathrm{sign}(\langle\Gamma_1, \Gamma_2\rangle) - \mathrm{sign}([\hat{B}^{\mathsf{T}}\Omega_2\hat{B}]_{12})\right)\right), \quad \text{(C.11)}$$

with $\Gamma_i = B\pi_i B^{-1}\Gamma = \hat{B}\pi_i\hat{B}^{-1}\Gamma$.

## C.2 $\Gamma_0(N)$ orbits of splittings

For CHL orbifolds the charge quantization condition $B\pi_i B^{-1}\Gamma \in \Lambda_m^* \oplus \Lambda_m$ for the splitting (C.6) does not reduce to a single condition. They will depend on the charge orbit, as well as on its twistedness, and only if $\gamma \in \mathbb{Z}_2 \ltimes \Gamma_0(N) \subset GL(2, \mathbb{Z})$, the quantization condition $B\pi_i B^{-1}\Gamma \in \Lambda_m^* \oplus \Lambda_m$ reduces to $\frac{aP-cQ}{k'} \in \Lambda_m^*$. Therefore it will be more convenient to decompose $M_2(\mathbb{Z})/\mathrm{Stab}(\pi_i)$ into orbits of $\gamma \in \Gamma_0(N)/\mathbb{Z}_2$ acting on the left.[33] Therefore we choose to decompose the splitting matrix as

$$\begin{pmatrix} p & q \\ r & s \end{pmatrix} = \begin{pmatrix} a & b \\ c & d \end{pmatrix}\cdot\begin{pmatrix} p' & j \\ 0 & k \end{pmatrix}, \quad \begin{pmatrix} a & b \\ c & d \end{pmatrix} \in \mathbb{Z}_2 \ltimes \Gamma_0(N), \quad p' > 0, \quad 0 \leq j < k, \quad \text{(C.12)}$$

---

[33]Note that $\mathrm{Dih}_4 \cap \mathbb{Z}_2 \ltimes \Gamma_0(N) = \mathbb{Z}_2 \times \mathbb{Z}_2$ and the corresponding quotient $\mathbb{Z}_2 \ltimes \Gamma_0(N)/[\mathbb{Z}_2 \times \mathbb{Z}_2] = \Gamma_0(N)/\mathbb{Z}_2$.

if $\frac{(p,r)}{\gcd(p,r)} = (*, 0) \bmod N$, and

$$\begin{pmatrix} p & q \\ r & s \end{pmatrix} = \begin{pmatrix} a & b \\ c & d \end{pmatrix} \cdot \begin{pmatrix} 0 & k \\ p' & j \end{pmatrix}, \quad \begin{pmatrix} a & b \\ c & d \end{pmatrix} \in \mathbb{Z}_2 \ltimes \Gamma_0(N), \quad p' > 0, \quad 0 \le j < Nk \tag{C.13}$$

otherwise. In the former case the splitting can be rotated under $\Gamma_0(N)$ to the canonical splitting (C.7), such that $\Gamma_1$ is in the $\Gamma_0(N)$ orbit of a purely electric charge. In this case we say that $\Gamma_1$ is of electric type and we call (C.7) 'splitting of electric type'. This splitting exists if and only if $P/k' \in \Lambda_m^*$. In contrast, the splitting (C.13) can be rotated under $\Gamma_0(N)$ to the canonical form

$$\begin{pmatrix} Q \\ P \end{pmatrix} = \begin{pmatrix} 0 \\ 1 \end{pmatrix}\left(P - \frac{j'}{k'}Q\right) + \begin{pmatrix} k' \\ j' \end{pmatrix}\frac{1}{k'}Q, \tag{C.14}$$

such that $\Gamma_1$ is in the $\Gamma_0(N)$ orbit of a purely magnetic charge. We then way $\Gamma_1$ is of magnetic type and we call (C.14) a 'splitting of magnetic type'. This splitting exists if and only if $Q/k' \in \Lambda_m$. Note that the second charge $\Gamma_2$ can be either of electric or of magnetic type in both types of splitting. In fact, we shall see that a splitting of mixed type, such that one charge is of electric type and the other of magnetic type, can be rotated by a suitable $\gamma \in \Gamma_0(N)$ into either type of splittings.

We drop the primes on $(j', k')$ in this discussion to simplify the notation, with the understanding that $k$ and $j$ are now relative prime. In the electric type, a splitting matrix with $k = 0 \bmod N$, such that $\binom{j}{k}\frac{1}{k}P$ is of electric type, can be rotated by a $\Gamma_0(N)$ element to another splitting of electric type

$$\begin{pmatrix} 1 & j \\ 0 & k \end{pmatrix} = \begin{pmatrix} j & b \\ k & -\tilde{j} \end{pmatrix}\begin{pmatrix} \tilde{j} & 1 \\ k & 0 \end{pmatrix}, \tag{C.15}$$

with $0 \le \tilde{j} < k$, $j\tilde{j} + bk = 1$. In the case where $k \ne 0 \bmod N$, such that $\binom{j}{k}\frac{1}{k}P$ is of magnetic type, an element of $\Gamma_0(N)$ rotates it to a splitting of magnetic type

$$\begin{pmatrix} 1 & j \\ 0 & k \end{pmatrix} = \begin{pmatrix} a & j \\ -\tilde{j} & k \end{pmatrix}\begin{pmatrix} k & 0 \\ \tilde{j} & 1 \end{pmatrix}, \tag{C.16}$$

with $\tilde{j} = 0 \bmod N$, $\tilde{j} < Nk$. This can be understood as follows: in (C.15), the second charge in the splitting is also electric since $k = 0 \bmod N$, and thus exchanging $(Q_1, P_1)$ with $(Q_2, P_2)$ preserves the type of the splitting; in (C.16), the second charge is magnetic since $k \ne 0 \bmod N$, and thus exchanging the two charges of the splitting sends the splitting of electric type to a splitting of magnetic type. The same reasoning applies to the splitting of magnetic types: when $j = 0 \bmod N$, such that $\binom{k}{j}\frac{1}{k}Q$ is of electric type, one has

$$\begin{pmatrix} 0 & k \\ 1 & j \end{pmatrix} = \begin{pmatrix} k & -\tilde{j} \\ j & d \end{pmatrix}\begin{pmatrix} \tilde{j} & 1 \\ k & 0 \end{pmatrix}, \tag{C.17}$$

with $d \ne 0 \bmod N$, $0 \le \tilde{j} < k$, and when $j \ne 0 \bmod N$, such that $\binom{k}{j}\frac{1}{k}Q$ is of magnetic type,

$$\begin{pmatrix} 0 & k \\ 1 & j \end{pmatrix} = \begin{pmatrix} -\tilde{j} & k \\ c & j \end{pmatrix}\begin{pmatrix} k & 0 \\ \tilde{j} & 1 \end{pmatrix}, \tag{C.18}$$

with $c = 0 \bmod N$, $0 \le \tilde{j} < Nk$ and $j\tilde{j} + ck = -1$.

It follows from this discussion that the splittings are in one-to-one correspondence with the cosets

$$M_2(\mathbb{Z})/\text{Stab}(\pi_i) = \left\{ \gamma \cdot \begin{pmatrix} 1 & j' \\ 0 & k' \end{pmatrix}, \ \gamma \in \Gamma_0(N)/\mathbb{Z}_2, \ 0 \le j' < k', \ (j', k') = 1 \right\} \tag{C.19}$$

$$\cup \left\{ \gamma \cdot \begin{pmatrix} 0 & k' \\ 1 & j' \end{pmatrix}, \ \gamma \in \Gamma_0(N)/\mathbb{Z}_2, \ 0 \le j' < Nk', \ j' \ne 0 \bmod N, \ (j', k') = 1 \right\}$$

$$= \left\{ \gamma \cdot \begin{pmatrix} 1 & j' \\ 0 & k' \end{pmatrix}, \ \gamma \in \Gamma_0(N)/\mathbb{Z}_2, \ 0 \le j' < k', \ k' \ne 0 \bmod N, \ (j', k') = 1 \right\}$$

$$\cup \left\{ \gamma \cdot \begin{pmatrix} 0 & k' \\ 1 & j' \end{pmatrix}, \ \gamma \in \Gamma_0(N)/\mathbb{Z}_2, \ 0 \le j' < Nk', \ (j', k') = 1 \right\},$$

where the splittings of mixed type are included either in the electric type or the magnetic type. In the following we shall consider both representatives, keeping in mind that we systematically double-count the splittings of mixed type in this way.

It is worth noting that the sign $(-1)^{\langle \Gamma_1, \Gamma_2 \rangle}$ appearing in the wall-crossing formula (2.12) does not depend on the type of splitting. For an electric-type splitting

$$\langle \Gamma_1, \Gamma_2 \rangle = (Q - \tfrac{j'}{k'} P) \cdot P = Q \cdot P \bmod 2 \, . \tag{C.20}$$

To prove this, note that either $P \notin N\Lambda^*$ and $\frac{1}{k'} P \in \Lambda$ so $(\frac{1}{k'} P)^2 = 0 \bmod 2$, or $P \in N\Lambda^*$ and $\frac{1}{k'} P \in \Lambda^*$ so $(\frac{1}{k'} P) \cdot P = 0 \bmod 2$. The same reasoning shows for a magnetic-type splitting

$$\langle \Gamma_1, \Gamma_2 \rangle = Q \cdot (P - \tfrac{j'}{k'} Q) = Q \cdot P \bmod 2 \, . \tag{C.21}$$

Moreover, under $\Gamma_0(N)$ the parity of $Q \cdot P$ is preserved:

$$(aQ + bP) \cdot (cQ + dP) = Q \cdot P + ac \, Q^2 + 2bc \, Q \cdot P + bd \, P^2 = Q \cdot P \bmod 2 \, . \tag{C.22}$$

## C.3  Factorization of the measure factor

We now discuss the factorization of the measure factor associated to the poles of $1/\Phi_{k-2}$ and $1/\tilde{\Phi}_{k-2}$ for $|\Omega_2| > \frac{1}{4}$ displayed in (5.75). In this subsection we show that whenever a term in the measure associated to the charge $\Gamma$ factorizes, it produces the correct measure factor of the corresponding 1/2-BPS charges $\Gamma_i$.

• For the first term in (5.75), we combine the sum over $A \in M_2(\mathbb{Z})/GL(2, \mathbb{Z})$ and the sum over $\gamma \in GL(2, \mathbb{Z})/\mathrm{Dih}_4$ in (5.57) as in (C.9), and use the decomposition (C.10) to get

$$\sum_{\substack{A \in M_2(\mathbb{Z})/GL(2,\mathbb{Z}) \\ A^{-1}\binom{Q}{P} \in \Lambda_m \oplus \Lambda_m}} |A| \Big( C_{k-2}\Big[A^{-1}\big(\begin{smallmatrix} -Q^2 & -Q \cdot P \\ -Q \cdot P & -P^2 \end{smallmatrix}\big)A^{-\intercal}; A^{\intercal}\Omega_2 A\Big] - C_{k-2}^F\Big[A^{-1}\big(\begin{smallmatrix} -Q^2 & -Q \cdot P \\ -Q \cdot P & -P^2 \end{smallmatrix}\big)A^{-\intercal}\Big] \Big)$$

$$= \sum_{\substack{A \in M_2(\mathbb{Z})/\mathrm{Dih}_4 \\ A^{-1}\Gamma \in \Lambda_m \oplus \Lambda_m}} |A| c_k\big(-\tfrac{([A^{-1}\Gamma]_1)^2}{2}\big) c_k\big(-\tfrac{([A^{-1}\Gamma]_2)^2}{2}\big)$$

$$\times \Big(-\frac{\delta([A^{\intercal}\Omega_2 A]_{12})}{4\pi} + \frac{[A^{-1}\Gamma]_1 \cdot [A^{-1}\Gamma]_2}{2}\big(\mathrm{sign}([A^{-1}\Gamma]_1 \cdot [A^{-1}\Gamma]_2) - \mathrm{sign}([A^{\intercal}\Omega_2 A]_{12})\big)\Big)$$

$$= \sum_{\substack{B \in M_2(\mathbb{Z})/\mathrm{Dih}_4 \\ \hat{B}^{-1}\Gamma \in \Lambda_m \oplus \Lambda_m}} \sum_{\substack{d_1 \geq 1 \\ \Gamma_1/d_1 \in \Lambda_m \oplus \Lambda_m}} c_k\big(-\tfrac{\gcd(Q_1^2, P_1^2, Q_1 \cdot P_1)}{2d_1^2}\big) \sum_{\substack{d_2 \geq 1 \\ \Gamma_2/d_2 \in \Lambda_m \oplus \Lambda_m}} c_k\big(-\tfrac{\gcd(Q_2^2, P_2^2, Q_2 \cdot P_2)}{2d_2^2}\big)$$

$$\times \Big(-\frac{\delta([\hat{B}^{\intercal}\Omega_2 \hat{B}]_{12})}{4\pi} + \frac{\langle \Gamma_1, \Gamma_2 \rangle}{2}\big(\mathrm{sign}(\langle \Gamma_1, \Gamma_2 \rangle) - \mathrm{sign}([\hat{B}^{\intercal}\Omega_2 \hat{B}]_{12})\big)\Big) \, , \tag{C.23}$$

where $B$ determines a splitting $\Gamma = \Gamma_1 + \Gamma_2$. In this sum, the only non-trivial contributions arise when $\Gamma_1$ is of electric type, such that $\gcd(Q_1^2, P_1^2, Q_1 P_1) = \frac{\gcd(NQ_1^2, P_1^2, Q_1 P_1)}{N}$, and because it is electric in $\Lambda_m$, $\Gamma_1/d_1$ is untwisted. Whereas, when $\Gamma_1/d_1$ is of magnetic type, $\gcd(Q_1^2, P_1^2, Q_1 P_1) = \gcd(NQ_1^2, P_1^2, Q_1 P_1)$, and because it is magnetic in $\Lambda_m$, $\Gamma_1$ can be either twisted or untwisted. Therefore we get the correct contribution to the measure for 1/2-BPS displayed in (2.22).

• For the third term in the measure (5.75), it is convenient to consider instead

$\tilde{A} = \begin{pmatrix} 1 & 0 \\ 0 & N \end{pmatrix} A \in M_{2,00}(N)$ such that

$$\sum_{\substack{A \in M_2(\mathbb{Z})/GL(2,\mathbb{Z}) \\ A^{-1}\binom{Q}{P/N} \in \Lambda_m^* \oplus \Lambda_m^*}} |A| \left( C_{k-2}\left[ A^{-1}\begin{pmatrix} -NQ^2 & -Q \cdot P \\ -Q \cdot P & -P^2/N \end{pmatrix} A^{-\intercal}; A^{\intercal}\Omega_2 A \right] - C_{k-2}^F\left[ A^{-1}\begin{pmatrix} -NQ^2 & -Q \cdot P \\ -Q \cdot P & -P^2/N \end{pmatrix} A^{-\intercal} \right] \right)$$

$$= \sum_{\substack{\tilde{A} \in M_{2,00}(N)/\mathrm{Dih}_4 \\ \tilde{A}^{-1}\Gamma \in \Lambda_m^* \oplus \Lambda_m^*}} |\tilde{A}| c_k\left( -N \frac{([\tilde{A}^{-1}\Gamma]_1)^2}{2} \right) c_k\left( -N \frac{([\tilde{A}^{-1}\Gamma]_2)^2}{2} \right)$$

$$\times \left( -\frac{\delta([\tilde{A}^{\intercal}\Omega_2 \tilde{A}]_{12})}{4\pi} + \frac{[\tilde{A}^{-1}\Gamma]_1 \cdot [\tilde{A}^{-1}\Gamma]_2}{2}\left( \mathrm{sign}([\tilde{A}^{-1}\Gamma]_1 \cdot [\tilde{A}^{-1}\Gamma]_2) - \mathrm{sign}([\tilde{A}^{\intercal}\Omega_2 \tilde{A}]_{12}) \right) \right). \quad (C.24)$$

A matrix $\tilde{A} \in M_{2,00}(N)$ admits either a decomposition with $\gamma \in \Gamma_0(N)$ such that

$$\tilde{A} = \gamma \cdot \begin{pmatrix} p' & j \\ 0 & Nk \end{pmatrix} = \gamma \cdot \begin{pmatrix} 1 & \frac{j}{Nk} \\ 0 & 1 \end{pmatrix}\begin{pmatrix} p' & 0 \\ 0 & Nk \end{pmatrix}, \quad (C.25)$$

and $\Gamma_\gamma = \gamma^{-1}\Gamma$ satisfies

$$\frac{P_\gamma}{k} \in N\Lambda_m^*, \qquad \frac{Q_\gamma - \frac{j}{kN}P_\gamma}{p'} \in \Lambda_m^*, \quad (C.26)$$

or a decomposition with $\gamma \in \Gamma_0(N)$ such that

$$\tilde{A} = \gamma \cdot \begin{pmatrix} 0 & k \\ Np' & Nj \end{pmatrix} = \gamma \cdot \begin{pmatrix} 0 & 1 \\ 1 & \frac{Nj}{k} \end{pmatrix}\begin{pmatrix} Np' & 0 \\ 0 & k \end{pmatrix}, \quad (C.27)$$

and

$$\frac{Q_\gamma}{k} \in \Lambda_m^*, \qquad \frac{P_\gamma - \frac{Nj}{k}Q_\gamma}{p'} \in N\Lambda_m^*. \quad (C.28)$$

For the splitting matrix of electric type (C.25), the charge $\Gamma_1$ is of electric type with

$$N([\tilde{A}^{-1}\Gamma]_1)^2 = \frac{\gcd(NQ_1^2, P_1^2, Q_1 \cdot P_1)}{p'^2}, \quad (C.29)$$

with the divisor integer $d_1 = p'$; and either the second charge $\Gamma_2$ is of electric type, with $\frac{kN}{\gcd(j,kN)} = 0 \bmod N$ and

$$N([\tilde{A}^{-1}\Gamma]_2)^2 = \frac{\gcd(NQ_2^2, P_2^2, Q_2 P_2)}{\gcd(j, Nk)^2}, \quad (C.30)$$

with the divisor integer $d_2 = \gcd(j, Nk)$, or $\Gamma_2$ is of twisted magnetic type with $\frac{kN}{\gcd(j,kN)} \neq 0 \bmod N$ and

$$N([\tilde{A}^{-1}\Gamma]_2)^2 = \frac{\gcd(NQ_2^2, P_2^2, Q_2 P_2)}{N(\gcd(j, Nk)/N)^2}, \quad (C.31)$$

with the divisor integer $d_2 = \gcd(j, Nk)/N$.

For the splitting matrix of magnetic type (C.27) the first charge $\Gamma_1$ is of untwisted magnetic type with

$$N([\tilde{A}^{-1}\Gamma]_1)^2 = \frac{\gcd(NQ_1^2, P_1^2, Q_1 \cdot P_1)}{Np'^2}, \quad (C.32)$$

with the divisor integer $d_1 = p'$; and either the second charge $\Gamma_2$ is of electric type, with $\frac{Nj}{\gcd(Nj,k)} = 0 \bmod N$ and

$$N([\tilde{A}^{-1}\Gamma]_2)^2 = \frac{\gcd(NQ_2^2, P_2^2, Q_2 P_2)}{\gcd(Nj, k)^2}, \quad (C.33)$$

with the divisor integer $d_2 = \gcd(Nj, k)$, or $\Gamma_2$ is of twisted magnetic type with $\frac{Nj}{\gcd(Nj,k)} \neq 0$ mod $N$ and

$$N([\tilde{A}^{-1}\Gamma]_2)^2 = \frac{\gcd(NQ_2^2, P_2^2, Q_2 P_2)}{N(\gcd(Nj,k)/N)^2} \,, \tag{C.34}$$

with the divisor integer $d_2 = \gcd(Nj, k)/N$.

- At last we consider the second term in (5.75), which is a combination . We combine the sum over $A \in M_{2,0}(\mathbb{Z})/[\mathbb{Z}_2 \ltimes \Gamma_0(N)]$ and the sum over $\gamma \in \Gamma_0(N)/\mathbb{Z}_2$ in (5.58) to get

$$\sum_{\substack{A\in M_{2,0}(N)/[\mathbb{Z}_2\ltimes\Gamma_0(N)] \\ A^{-1}\binom{Q}{P}\in\Lambda_m^*\oplus\Lambda_m}} |A| \left( \widetilde{C}_{k-2}\left[A^{-1}\begin{pmatrix} -Q^2 & -Q\cdot P \\ -Q\cdot P & -P^2 \end{pmatrix}A^{-\intercal}; A^{\intercal}\Omega_2 A\right] - \widetilde{C}_{k-2}^F\left[A^{-1}\begin{pmatrix} -Q^2 & -Q\cdot P \\ -Q\cdot P & -P^2 \end{pmatrix}A^{-\intercal}\right]\right)$$

$$+ \sum_{\substack{A\in M_{2,0}(N)/[\mathbb{Z}_2\times\mathbb{Z}_2] \\ A^{-1}\Gamma\in\Lambda_m^*\oplus\Lambda_m}} |A| c_k\left(-N\frac{([A^{-1}\Gamma]_1)^2}{2}\right) c_k\left(-\frac{([A^{-1}\Gamma]_2)^2}{2}\right) \tag{C.35}$$

$$\times \left( -\frac{\delta([A^{\intercal}\Omega_2 A]_{12})}{4\pi} + \frac{[A^{-1}\Gamma]_1 \cdot [A^{-1}\Gamma]_2}{2}\left(\text{sign}([A^{-1}\Gamma]_1 \cdot [A^{-1}\Gamma]_2) - \text{sign}([A^{\intercal}\Omega_2 A]_{12})\right)\right).$$

A matrix in $A \in M_{2,0}(N)$ admits one of the following decompositions with respect to $\gamma \in \Gamma_0(N)$:

1.

$$A = \gamma \cdot \begin{pmatrix} 1 & \frac{j}{k} \\ 0 & 1 \end{pmatrix}\begin{pmatrix} p' & 0 \\ 0 & k \end{pmatrix} \quad \Rightarrow \quad \frac{Q_\gamma - \frac{j}{k}P_\gamma}{p'} \in \Lambda_m^* \,, \quad \frac{P_\gamma}{k} \in \Lambda_m \,, \tag{C.36}$$

$\Gamma_1$ is always of electric type and $N([A^{-1}\Gamma]_1)^2 = \frac{\gcd(NQ_1^2, P_1^2, Q_1 P_1)}{p'^2}$, and $\Gamma_2$ is either of untwisted electric type with $\frac{k}{\gcd(j,k)} = 0$ mod $N$ with $([A^{-1}\Gamma]_2)^2 = \frac{\gcd(NQ_2^2, P_2^2, Q_2 P_2)}{N\gcd(j,k)^2}$ or of magnetic type with $\frac{k}{\gcd(j,k)} \neq 0$ mod $N$ with $([A^{-1}\Gamma]_2)^2 = \frac{\gcd(NQ_2^2, P_2^2, Q_2 P_2)}{\gcd(j,k)^2}$.

2.

$$A = \gamma \cdot \begin{pmatrix} 0 & 1 \\ N & \frac{j}{k} \end{pmatrix}\begin{pmatrix} p' & 0 \\ 0 & k \end{pmatrix} \quad \Rightarrow \quad \frac{P_\gamma - \frac{j}{k}Q_\gamma}{p'} \in N\Lambda_m^* \,, \quad \frac{Q_\gamma}{k} \in \Lambda_m \,, \tag{C.37}$$

$\Gamma_1$ is always of untwisted magnetic type and $N([A^{-1}\Gamma]_1)^2 = \frac{\gcd(NQ_1^2, P_1^2, Q_1 P_1)}{Np'^2}$, and $\Gamma_2$ is either of untwisted electric type with $\frac{j}{\gcd(j,k)} = 0$ mod $N$ with $([A^{-1}\Gamma]_2)^2 = \frac{\gcd(NQ_2^2, P_2^2, Q_2 P_2)}{N\gcd(j,k)^2}$ or of magnetic type with $\frac{j}{\gcd(j,k)} \neq 0$ mod $N$ with $([A^{-1}\Gamma]_2)^2 = \frac{\gcd(NQ_2^2, P_2^2, Q_2 P_2)}{\gcd(j,k)^2}$.

3.

$$A = \gamma \cdot \begin{pmatrix} 1 & \frac{j}{Nk} \\ 0 & 1 \end{pmatrix}\begin{pmatrix} 0 & p' \\ Nk & 0 \end{pmatrix} \quad \Rightarrow \quad \frac{Q_\gamma - \frac{j}{Nk}P_\gamma}{p'} \in \Lambda_m \,, \quad \frac{P_\gamma}{k} \in N\Lambda_m^* \,, \tag{C.38}$$

$\Gamma_2$ is always of untwisted electric type and $([A^{-1}\Gamma]_2)^2 = \frac{\gcd(NQ_2^2, P_2^2, Q_2 P_2)}{Np'^2}$, and $\Gamma_1$ is either of electric type with $\frac{Nk}{\gcd(j,Nk)} = 0$ mod $N$ with $N([A^{-1}\Gamma]_1)^2 = \frac{\gcd(NQ_1^2, P_1^2, Q_1 P_1)}{\gcd(j,Nk)^2}$ or of untwisted magnetic type with $\frac{Nk}{\gcd(j,Nk)} \neq 0$ mod $N$ with $N([A^{-1}\Gamma]_1)^2 = \frac{\gcd(NQ_1^2, P_1^2, Q_1 P_1)}{N(\gcd(j,Nk)/N)^2}$.

4.

$$A = \gamma \cdot \begin{pmatrix} 0 & 1 \\ 1 & \frac{Nj}{k} \end{pmatrix} \begin{pmatrix} 0 & p' \\ k & 0 \end{pmatrix} \quad \Rightarrow \quad \frac{P_\gamma - \frac{Nj}{k} Q_\gamma}{p'} \in \Lambda_m, \quad \frac{Q_\gamma}{k} \in \Lambda_m^*, \tag{C.39}$$

$\Gamma_2$ is always of magnetic type and $([A^{-1}\Gamma]_2)^2 = \frac{\gcd(NQ_2^2, P_2^2, Q_2 P_2)}{p'^2}$, and $\Gamma_1$ is either of electric type with $\frac{Nj}{\gcd(Nj,k)} = 0 \mod N$ with $N([A^{-1}\Gamma]_1)^2 = \frac{\gcd(NQ_1^2, P_1^2, Q_1 P_1)}{\gcd(Nj,k)^2}$ or of untwisted magnetic type with $\frac{Nj}{\gcd(Nj,k)} \neq 0 \mod N$ with $N([A^{-1}\Gamma]_1)^2 = \frac{\gcd(NQ_1^2, P_1^2, Q_1 P_1)}{N(\gcd(Nj,k)/N)^2}$.

We conclude that after trading each of the sums over $A$ as sums over splitting matrices $B$, the contribution from (5.75) gives a term of the form

$$\left( -\frac{\delta([\hat{B}^\intercal \Omega_2 \hat{B}]_{12})}{4\pi} + \frac{\langle \Gamma_1, \Gamma_2 \rangle}{2} \left( \text{sign}(\langle \Gamma_1, \Gamma_2 \rangle) - \text{sign}([\hat{B}^\intercal \Omega_2 \hat{B}]_{12}) \right) \right) c'(\Gamma_1) c'(\Gamma_2) \tag{C.40}$$

to the last line in (5.92), where $c'(\Gamma_i)$ is either

$$c_U(\Gamma_i) = \sum_{\substack{d_i > 1 \\ d_i^{-1}\Gamma_i \in \Lambda_m \oplus N\Lambda_m^*}} c_k \left( -\frac{\gcd(NQ_i^2, P_i^2, Q_i \cdot P_i)}{2Nd_i^2} \right), \tag{C.41}$$

when the contribution is only non-vanishing for untwisted charge $\Gamma_i$, or

$$c_T(\Gamma_i) = \sum_{\substack{d_i > 1 \\ d_i^{-1}\Gamma_i \in \Lambda_m^* \oplus \Lambda_m}} c_k \left( -\frac{\gcd(NQ_i^2, P_i^2, Q_i \cdot P_i)}{2d_i^2} \right), \tag{C.42}$$

for generic contribution such the charge $\Gamma_i$ is either twisted or untwisted.

It remains to show that the three terms in the measure count all the possible splittings with the correct multiplicity, so as to reproduce the product of the summation factors of formula (2.22) for the two charges $\Gamma_i$.

## C.4 Electric-magnetic type of splittings

We summarize the conditions from the three terms in (5.75) to contribute to a given splitting in Table 2, where for the second term we distinguish the cases where $B^{-1}(Q, P) \in \Lambda_m^* \oplus \Lambda_m$ or $B^{-1}(Q, P) \in \Lambda_m \oplus \Lambda_m^*$.

For this purpose we enumerate the possible 1/4-BPS charges $\Gamma$ and the type of 1/2-BPS charges they can possibly split into, *i.e.* twisted or untwisted, electric or magnetic. It will be convenient to introduce some notation for classifying pairs of 1/2-BPS charges: for each type of splitting we define a 2-component vector which first component accounts for the electric type charges and the second for the magnetic type charges, with a $U$ for untwisted and a $T$ for twisted. *e.g.*

1. $(TT, \emptyset)$, $(T, T)$ and $(\emptyset, TT)$ stand for electric-twisted electric-twisted, electric-twisted magnetic-twisted, and magnetic-twisted magnetic-twisted splittings, respectively.

2. $(TU, \emptyset)$, $(T, U)$, $(U, T)$ and $(\emptyset, TU)$ stand for electric-twisted electric-untwisted, electric-twisted magnetic-untwisted, electric-untwisted magnetic-twisted and magnetic-twisted magnetic-untwisted splittings, respectively.

Table 2: $\Gamma_0(N)$ orbits of splittings from the three terms in (5.75). The first column indicates the support of $B^{-1}(Q, P)$. The second and third columns give the corresponding constraints on $\Gamma_1, \Gamma_2$, for each of the two possible splittings (C.7) and (C.14). The last column records the counting function. We write $k = Nk'$ and $j = Nj'$ whenever $k$ or $j$ are forced to be multiple of $N$. $\mathcal{O}_{ij}$ is used in the text to denote in the table above contribution from row $i$ and column $j$.

| $\mathcal{O}_{ij}$ | Electric type | | Magnetic type | | Counted by |
|---|---|---|---|---|---|
| $\Lambda_m \oplus \Lambda_m$ | $Q - \frac{j}{k}P \in \Lambda_m$, | $\frac{1}{k}P \in \Lambda_m$ | $P - \frac{j}{k}Q \in \Lambda_m$, | $\frac{1}{k}Q \in \Lambda_m$ | $\Phi_{k-2}^{-1}(\Omega)$ |
| $\Lambda_m^* \oplus \Lambda_m$ | $Q - \frac{j}{k}P \in \Lambda_m^*$, | $\frac{1}{k}P \in \Lambda_m$ | $P - \frac{j}{k}Q \in N\Lambda_m^*$, | $\frac{1}{k}Q \in \Lambda_m$ | $\tilde{\Phi}_{k-2}^{-1}(\Omega)$ |
| $\Lambda_m \oplus \Lambda_m^*$ | $Q - \frac{j}{Nk'}P \in \Lambda_m$, | $\frac{1}{Nk'}P \in \Lambda_m^*$ | $P - \frac{Nj'}{k}Q \in \Lambda_m$, | $\frac{1}{k}Q \in \Lambda_m^*$ | $\tilde{\Phi}_{k-2}^{-1}(\Omega)$ |
| $\Lambda_m^* \oplus \Lambda_m^*$ | $Q - \frac{j}{Nk'}P \in \Lambda_m^*$, | $\frac{1}{Nk'}P \in \Lambda_m^*$ | $P - \frac{Nj'}{k}Q \in N\Lambda_m^*$, | $\frac{1}{k}Q \in \Lambda_m^*$ | $\Phi_{k-2}^{-1}(\Omega/N)$ |

3. $(UU, \emptyset)$, $(U, U)$ and $(\emptyset, UU)$ stand for electric-untwisted electric-untwisted, electric-untwisted magnetic-untwisted, and magnetic-untwisted magnetic-untwisted splittings, respectively.

We shall enumerate the possible splittings according to the following graph of inclusions,

$$N\Lambda_e \oplus N\Lambda_m \quad \begin{matrix} \subset \\ \subset \end{matrix} \quad \begin{matrix} N\Lambda_e \oplus N\Lambda_e \\ \Lambda_m \oplus N\Lambda_m \end{matrix} \quad \begin{matrix} \subset \\ \subset \end{matrix} \quad \Lambda_m \oplus N\Lambda_e \quad \begin{matrix} \subset \\ \subset \end{matrix} \quad \begin{matrix} \Lambda_m \oplus \Lambda_m \\ \Lambda_e \oplus N\Lambda_e \end{matrix} \quad \begin{matrix} \subset \\ \subset \end{matrix} \quad \Lambda_e \oplus \Lambda_m . \quad \text{(C.43)}$$

We will denote $X \Subset \Lambda$ the strict inclusion of the vector $X$ in $\Lambda$, meaning that $X$ is a generic vector in $\Lambda$ and does not belong to a smaller lattice $\tilde{\Lambda}$ in this sequence

$$\ldots \subset N^k \Lambda_m \subset N^k \Lambda_m^* \subset \ldots \subset N\Lambda_m \subset N\Lambda_m^* \subset \Lambda_m \subset \Lambda_m^* . \quad \text{(C.44)}$$

In the following, it will be convenient to recall the generating function whose Fourier coefficients give the contribution to the measure. According to table 2, the factorizations (A.44) imply that when the condition is $\frac{1}{d_i}[B^{-1}\Gamma]_i \in \Lambda_m$, the corresponding measure factor for the $1/2$-BPS charge $B\pi_i B^{-1}\Gamma$ is a Fourier coefficient of $\Delta_k(\tau)^{-1}$, whereas when $\frac{1}{d_i}[B^{-1}\Gamma]_i \in \Lambda_m^*$ it is a Fourier coefficient of $\Delta_k(\tau/N)^{-1}$. For a magnetic type charge $B\pi_i B^{-1}\Gamma$, $\Delta_k(\tau)^{-1}$ gives a contribution $c_T(\Gamma)$ and $\Delta_k(\tau/N)^{-1}$ a contribution $c_U(\Gamma)$. On the contrary for an eletric type charge, $\Delta_k(\tau)^{-1}$ gives a contribution $c_U(\Gamma)$ and $\Delta_k(\tau/N)^{-1}$ a contribution $c_T(\Gamma)$.

There are seven cases of interest:

1. $Q \Subset \Lambda_m^*, P \Subset \Lambda_m$ : the only contributions are from $\mathcal{O}_{21}$ and $\mathcal{O}_{32}$. These two contributions give $(T, T)$ splittings with Fourier contributions in $[\Delta_k(\rho/N)\Delta_k(\sigma)]^{-1}$. We thus obtain a single contribution in $c_T(\Gamma_1)c_T(\Gamma_2)$ (with (C.42)), as expected for twisted $1/2$-BPS charges.

2. $Q \Subset \Lambda_m, P \Subset \Lambda_m$ : contributions from $\mathcal{O}_{11}, \mathcal{O}_{21}, \mathcal{O}_{12}, \mathcal{O}_{22}$, and $\mathcal{O}_{32}$ fall in $(U, T), (\emptyset, UT)$, and $(\emptyset, TT)$ splitting sectors.

   Electric-type splitting : the first charge in (C.7) is purely electric and thus untwisted in both $\mathcal{O}_{11}$ and $\mathcal{O}_{21}$, the second one is congruent to an electric charge for $k = 0 \bmod N$, and a magnetic one otherwise. But since $P \Subset \Lambda_m$, $\frac{1}{k}P \in \Lambda_m$ implies that $k \neq 0 \bmod N$, and thus the second charge in (C.7) is magnetic-twisted. $\mathcal{O}_{11}$ and $\mathcal{O}_{21}$ combine together

to give $(U, T)$ splittings with measure factor $(c_T(\Gamma_1) + c_U(\Gamma_1))c_T(\Gamma_2)$ coming from Fourier coefficients of $\left(\Delta_k(\rho)^{-1} + \Delta_k(\rho/N)^{-1}\right)\Delta_k(\sigma)^{-1}$.

Magnetic-type splitting : the first charge in (C.14) is purely magnetic, and thus twisted for $\mathcal{O}_{32}$, untwisted for $\mathcal{O}_{22}$, and can be either twisted or untwisted for $\mathcal{O}_{12}$, the second 1/2-BPS charge is congruent to a magnetic-untwisted charge for $j \neq 0 \bmod N$, and electric-twisted otherwise.

When $P - \frac{j}{k}Q \in N\Lambda_m^*$, $\mathcal{O}_{12}$ contribute only when $j \neq 0 \bmod N$, thus combining with $\mathcal{O}_{22}$ to give $(\emptyset, TU)$ splittings with the measure $(c_T(\Gamma_1) + c_U(\Gamma_1))c_T(\Gamma_2)$ coming from Fourier coefficients of $\left(\Delta_k(\rho)^{-1} + \Delta_k(\rho/N)^{-1}\right)\Delta_k(\sigma)^{-1}$.

When $P - \frac{j}{k}Q \subseteq \Lambda_m$ with $j \neq 0 \bmod N$, $\mathcal{O}_{12}$ gives $(\emptyset, TT)$ splittings with measure $c_T(\Gamma_1) c_T(\Gamma_2)$ from Fourier coefficients of $[\Delta_k(\rho)\Delta_k(\sigma)]^{-1}$. Finally, when $j = 0 \bmod N$, $\mathcal{O}_{12}$ combines with $\mathcal{O}_{32}$ — for which $j = 0 \bmod N$ by construction — to give $(U, T)$ splittings with measure $c_T(\Gamma_1)(c_T(\Gamma_2) + c_U(\Gamma_2))$ from Fourier coefficients of $\left(\Delta_k(\rho)^{-1} + \Delta_k(\rho/N)^{-1}\right) \Delta_k(\sigma)^{-1}$. Recall that we double-count this last splitting, which is the same as the one defined above from $\mathcal{O}_{11}$ and $\mathcal{O}_{21}$ with $\Gamma_1$ and $\Gamma_2$ exchanged, according to (C.19).

3. $Q \subseteq \Lambda_m^*, P \subseteq N\Lambda_m^*$ : contribution from $\mathcal{O}_{21}, \mathcal{O}_{31}, \mathcal{O}_{41}, \mathcal{O}_{32}$ and $\mathcal{O}_{42}$ fall in $(TT, \emptyset)$, $(TU, \emptyset)$, and $(T, U)$.

   Electric-type splitting : the contributions $\mathcal{O}_{31}, \mathcal{O}_{41}$ are both constrained to $k = 0 \bmod N$, imposing the second charge in (C.7) to be electric-twisted.

   If $j \neq 0 \bmod N$ and $Q - \frac{j}{k}P \subseteq \Lambda_m^*$, the splitting is $(TT, \emptyset)$ and only $\mathcal{O}_{41}$ contributes accordingly, with measure $c_T(\Gamma_1)c_T(\Gamma_2)$ from $[\Delta_k(\rho/N)\Delta_k(\sigma/N)]^{-1}$.

   When $Q - \frac{j}{k}P \in \Lambda_m$, the splitting is $(TU, \emptyset)$ and both $\mathcal{O}_{31}, \mathcal{O}_{41}$ contribute with measure $(c_T(\Gamma_1) + c_U(\Gamma_1))c_T(\Gamma_2)$ from Fourier coefficients of $\Delta_k(\rho/N)^{-1}\left(\Delta_k(\sigma)^{-1} + \Delta_k(\sigma/N)^{-1}\right)$.

   If instead $j = Nj'$, contributions from $\mathcal{O}_{41}$, whose condition rewrites $Q - \frac{j'}{k'}P \in \Lambda_m^*$, $\frac{1}{k'}P \in N\Lambda_m^*$, combine with $\mathcal{O}_{21}$ to $(T, U)$ splittings with measure $c_T(\Gamma_1)(c_T(\Gamma_2) + c_U(\Gamma_2))$ from Fourier coefficients of $\Delta_k(\rho/N)^{-1}\left(\Delta_k(\sigma)^{-1} + \Delta_k(\sigma/N)^{-1}\right)$ — note that their second 1/2-BPS charge in (C.7) is congruent to a magnetic-untwisted one since $P \subseteq N\Lambda_m^*$ implies $k' \neq 0 \bmod N$ in $\mathcal{O}_{41}$.

   Magnetic-type splitting : contributions from $\mathcal{O}_{32}, \mathcal{O}_{42}$ have $j = 0 \bmod N$ by construction, imposing their second 1/2-BPS charge in (C.14) to be congruent to an electric-twisted one, as well as $P - \frac{Nj'}{k}Q \in N\Lambda^*$, implying the first 1/2-BPS charge to be magnetic-untwisted for both of them. They thus combine to give $(T, U)$ splittings with measure $(c_T(\Gamma_1) + c_U(\Gamma_2))c_T(\Gamma_2)$ from Fourier coefficients in $[\Delta_k(\rho/N)\left(\Delta_k(\sigma) + \Delta_k(\sigma/N)\right)]^{-1}$. Recall that we couble-count this last splitting, which is the same as the one defined above from $\mathcal{O}_{41}$ and $\mathcal{O}_{21}$ with $\Gamma_1$ and $\Gamma_2$ exchanged, according to (C.19).

4. $Q \subseteq \Lambda_m, P \subseteq N\Lambda_m^*$ : contribution from $\mathcal{O}_{11}, \mathcal{O}_{21}, \mathcal{O}_{31}, \mathcal{O}_{41}, \mathcal{O}_{12}, \mathcal{O}_{22}, \mathcal{O}_{32}$, and $\mathcal{O}_{42}$ fall symmetrically in $(U, U)$, $(TT, \emptyset)$, and $(\emptyset, TT)$.

   Electric-type splitting : $P \subseteq N\Lambda_m^*$ imposes $k \neq 0 \bmod N$ for $\mathcal{O}_{11}, \mathcal{O}_{21}$, for which the conditions rewrite $\frac{1}{k'}P \in N\Lambda_m^*$, with $k' \neq 0 \bmod N$, thus implying that the second 1/2-BPS charge in (C.7) is congruent to magnetic-untwisted one.

   Given $j = 0 \bmod N$ in $\mathcal{O}_{31}, \mathcal{O}_{41}$, one can rewrite their conditions as $Q - \frac{j'}{k'}P \in \Lambda_m$ and $\frac{1}{k'}P \in N\Lambda_m^*$, and these combine with $\mathcal{O}_{11}, \mathcal{O}_{21}$ to give $(U, U)$ splittings with measure $(c_T(\Gamma_1) + c_U(\Gamma_1))(c_T(\Gamma_2) + c_U(\Gamma_2))$ from Fourier coefficients of all factors $\left(\Delta_k(\rho)^{-1} + \Delta_k(\rho/N)^{-1}\right)\left(\Delta_k(\sigma)^{-1} + \Delta_k(\sigma/N)^{-1}\right)$. These are the only contributions from $\mathcal{O}_{11}$ and $\mathcal{O}_{12}$, because $k \neq 0 \bmod N$.

For $j \neq 0 \bmod N$, one has $Q - \frac{j}{Nk'}P \subseteq \Lambda_m^*$, and $\mathcal{O}_{31}$ is empty while $\mathcal{O}_{41}$ contributes alone to $(TT, \emptyset)$ splittings with measure $c_T(\Gamma_1)c_T(\Gamma_2)$ from $[\Delta_k(\rho/N)\Delta_k(\sigma/N))]^{-1}$.

Magnetic-type splitting : in the case where $j = 0 \bmod N$, all $\mathcal{O}_{12}$, $\mathcal{O}_{22}$, $\mathcal{O}_{32}$ and $\mathcal{O}_{42}$ combine, with $k \neq 0 \bmod N$ for each, to give $(U, U)$ splittings measure from Fourier coefficients of $\left(\Delta_k(\rho)^{-1} + \Delta_k(\rho/N)^{-1}\right)\left(\Delta_k(\sigma)^{-1} + \Delta_k(\sigma/N)^{-1}\right)$, double-counting the electric type $(U, U)$ splittings describe above. For $j \neq 0 \bmod N$, and when $P - \frac{j}{k}Q \subseteq \Lambda_m$, $\mathcal{O}_{12}$ contribute alone to $(\emptyset, TT)$ splittings, with measure $c_T(\Gamma_1)c_T(\Gamma_2)$ from Fourier coefficients of $[\Delta_k(\rho)\Delta_k(\sigma)]^{-1}$.

5. $Q \subseteq \Lambda_m$, $P \subseteq N\Lambda_m$ : contributions from all $\mathcal{O}_{ij}$ fall in $(U, U)$, $(UU, \emptyset)$, and $(\emptyset, TT)$ splitting sectors.

Electric-type splitting : cases with $k \neq 0 \bmod N$ and $Q - \frac{j}{k}P \in \Lambda_m$ appear in $\mathcal{O}_{11}$ and $\mathcal{O}_{21}$, together with $\mathcal{O}_{31}$ and $\mathcal{O}_{41}$ when $j = 0 \bmod N$, corresponding to $(U, U)$ splittings with the generic measure from Fourier coefficients of $\left(\Delta_k(\rho)^{-1} + \Delta_k(\rho/N)^{-1}\right) \times \left(\Delta_k(\sigma)^{-1} + \Delta_k(\sigma/N)^{-1}\right)$.

When $k = 0 \bmod N$, cases with $j \neq 0 \bmod N$ get contributions from $\mathcal{O}_{11}$, $\mathcal{O}_{21}$, $\mathcal{O}_{31}$ and $\mathcal{O}_{41}$, corresponding to $(UU, \emptyset)$ splittings with the generic measure from Fourier coefficients of $\left(\Delta_k(\rho)^{-1} + \Delta_k(\rho/N)^{-1}\right)\left(\Delta_k(\sigma)^{-1} + \Delta_k(\sigma/N)^{-1}\right)$.

Magnetic-type splitting : we obtain that $k \neq 0 \bmod N$ in all cases. When $j \neq 0 \bmod N$, one has $P - \frac{j}{k}Q \subseteq \Lambda_m$ and only $\mathcal{O}_{12}$ contributes, giving $(\emptyset, TT)$ splittings with measure $c_T(\Gamma_1)c_T(\Gamma_2)$ from $[\Delta_k(\rho)\Delta_k(\sigma)]^{-1}$.

When $j = 0 \bmod N$, $P - \frac{j}{k}Q \in N\Lambda_m$ and $\mathcal{O}_{12}$, $\mathcal{O}_{22}$, $\mathcal{O}_{32}$, $\mathcal{O}_{42}$ contribute, giving $(U, U)$ splittings with the generic measure from Fourier coefficients of all four factors in $\left(\Delta_k(\rho)^{-1} + \Delta_k(\rho/N)^{-1}\right)\left(\Delta_k(\sigma)^{-1} + \Delta_k(\sigma/N)^{-1}\right)$. These splittings are the same as the electric type splittings of the same $(U, U)$ type.

6. $Q \subseteq N\Lambda_m^*$, $P \subseteq N\Lambda_m^*$: all $\mathcal{O}_{ij}$ contribute and fall in $(U, U)$, $(\emptyset, UU)$, and $(TT, \emptyset)$ splitting sectors.

Electric-type splitting : when $k \neq 0 \bmod N$, $Q - \frac{j}{k}P \in N\Lambda_m^*$ and $\mathcal{O}_{11}$, $\mathcal{O}_{21}$, together with $\mathcal{O}_{31}$ and $\mathcal{O}_{41}$ when $j = 0 \bmod N$, contribute to $(U, U)$ splittings, with measure contributions from Fourier coefficients of all four factors in $\left(\Delta_k(\rho)^{-1} + \Delta_k(\rho/N)^{-1}\right) \times \left(\Delta_k(\sigma)^{-1} + \Delta_k(\sigma/N)^{-1}\right)$.

When $k = 0 \bmod N$, only the two last orbits can contribute, and when $j \neq 0 \bmod N$ there is no other contribution than $\mathcal{O}_{41}$, leading to $(TT, \emptyset)$ splittings with measure $c_T(\Gamma_1)c_T(\Gamma_2)$ from Fourier coefficients of $[\Delta_k(\rho/N)\Delta_k(\sigma/N)]^{-1}$.

Magnetic-type splitting : when $k \neq 0 \bmod N$, $\mathcal{O}_{12}$ and $\mathcal{O}_{22}$ with $j \neq 0 \bmod N$ contribute, together with $\mathcal{O}_{32}$ and $\mathcal{O}_{42}$ when $k = 0 \bmod N$, to $(\emptyset, UU)$ splittings with the generic measure from Fourier coefficients of $\left(\Delta_k(\rho)^{-1} + \Delta_k(\rho/N)^{-1}\right)\left(\Delta_k(\sigma)^{-1} + \Delta_k(\sigma/N)^{-1}\right)$.

When $k \neq 0 \bmod N$ and $j = 0 \bmod N$, $\mathcal{O}_{12}$, $\mathcal{O}_{22}$, $\mathcal{O}_{32}$ and $\mathcal{O}_{42}$ contribute to $(U, U)$ splittings with the generic measure from Fourier coefficients of $\left(\Delta_k(\rho)^{-1} + \Delta_k(\rho/N)^{-1}\right) \times \left(\Delta_k(\sigma)^{-1} + \Delta_k(\sigma/N)^{-1}\right)$, associated to the same splittings of electric type $(U, U)$ described above.

7. $Q \subseteq N\Lambda_m^*$, $P \subseteq N\Lambda_m$: all $\mathcal{O}_{ij}$ contribute and fall in $(U, U)$, $(\emptyset, UU)$, and $(UU, \emptyset)$ splitting sectors.

Electric-type splitting : when $k \neq 0 \bmod N$, $Q - \frac{j}{k}P \in N\Lambda_m^*$ and $\mathcal{O}_{11}$, $\mathcal{O}_{21}$, together with $\mathcal{O}_{31}$ and $\mathcal{O}_{41}$ when $j = 0 \bmod N$, contribute to $(U, U)$ splittings, with the generic measure from Fourier coefficients of $\left(\Delta_k(\rho)^{-1} + \Delta_k(\rho/N)^{-1}\right)\left(\Delta_k(\sigma)^{-1} + \Delta_k(\sigma/N)^{-1}\right)$.

When $k = 0 \bmod N$, all the four orbits can contribute and $j \neq 0 \bmod N$. They lead to $(UU, \emptyset)$ splittings with generic measure from Fourier coefficients of $\left(\Delta_k(\rho)^{-1} + \Delta_k(\rho/N)^{-1}\right)\left(\Delta_k(\sigma)^{-1} + \Delta_k(\sigma/N)^{-1}\right)$.

Magnetic-type splitting : when $k \neq 0 \bmod N$, $\mathcal{O}_{12}$ and $\mathcal{O}_{22}$ with $j \neq 0 \bmod N$ contribute, together with $\mathcal{O}_{32}$ and $\mathcal{O}_{42}$ when $k = 0 \bmod N$, to $(\emptyset, UU)$ splittings with the generic measure from Fourier coefficients of $\left(\Delta_k(\rho)^{-1} + \Delta_k(\rho/N)^{-1}\right)\left(\Delta_k(\sigma)^{-1} + \Delta_k(\sigma/N)^{-1}\right)$.

When $k \neq 0 \bmod N$ and $j = 0 \bmod N$, $\mathcal{O}_{12}$, $\mathcal{O}_{22}$, $\mathcal{O}_{32}$ and $\mathcal{O}_{42}$ contribute to $(U, U)$ splittings with the generic measure $(c_T(\Gamma_1) + c_U(\Gamma_1))(c_T(\Gamma_2) + c_U(\Gamma_2))$ from Fourier coefficients of $\left(\Delta_k(\rho)^{-1} + \Delta_k(\rho/N)^{-1}\right)\left(\Delta_k(\sigma)^{-1} + \Delta_k(\sigma/N)^{-1}\right)$, which count the same splitting of electric type described above.

This concludes the proof of formula (5.92). As a consistency check, we note that these results are consistent with Fricke duality. Namely, for 1/4-BPS charges belonging to Fricke-invariant subsets, such as $(Q, P) \underline{\in} \Lambda_m^* \oplus \Lambda_m$ or $(Q, P) \underline{\in} \Lambda_m \oplus N\Lambda_m^*$, the possible splittings are invariant under the exchange of electric and magnetic type; whereas for charges in subsets that are exchanged under Fricke duality, as $(Q, P) \underline{\in} \Lambda_m \oplus N\Lambda_m$ and $(Q, P) \underline{\in} N\Lambda^* \oplus N\Lambda^*$, the possible splittings are themselves exchanged under Fricke duality. Moreover, we find that all the splittings of electric-magnetic type are correctly double-counted through the splitting matrices of electric and magnetic type, consistently with (C.19).

# D  Two-instanton singular contributions to Abelian Fourier coefficients

In this section, we extract the contributions to the rank-2 Abelian Fourier modes from the Dirac delta functions in the Poincaré series representation (5.25), (5.57) of the Fourier coefficients of $\frac{1}{\Phi_{k-2}}$.

## D.1  Maximal rank

Starting from (5.24), the sum over $\gamma \in GL(2, \mathbb{Z})/\mathrm{Dih}_4$ can be unfolded against the integration domain,[34] by changing variables as $\Omega_2 \to \gamma^{-\intercal}\Omega_2\gamma^{-1}$. The contribution of the delta functions in (5.25) then leads to

$$
-\frac{R^4}{2\pi} \sum_{\substack{\widetilde{Q} \in \Lambda_{p-2,q-2}^{\oplus 2} \\ \gamma \in GL(2,\mathbb{Z})/\mathrm{Dih}_4}} \sum_{A \in M_2(\mathbb{Z})/GL(2,\mathbb{Z})} e^{2\pi i a^{iI}A_{ij}Q_I^j} \int_{\mathcal{P}_2} \frac{d^3\Omega_2}{\Omega_2^{3/2}} |\Omega_2|^{\frac{q-5}{2}} c\left(-\frac{(s\widetilde{Q}_1 - q\widetilde{Q}_2)^2}{2}\right) c\left(-\frac{(p\widetilde{Q}_2 - r\widetilde{Q}_1)^2}{2}\right)
$$

$$
\times \delta\left(\mathrm{tr}\begin{pmatrix} 0 & 1/2 \\ 1/2 & 0 \end{pmatrix}\Omega_2\right) e^{-\pi\mathrm{Tr}\left[\frac{R^2}{S_2}\Omega_2^{-1}\gamma^\intercal A^\intercal \begin{pmatrix} 1 & S_1 \\ S_1 & |S|^2 \end{pmatrix}A\gamma + 2\Omega_2\gamma^{-1}\widetilde{Q}\cdot\widetilde{Q}^\intercal\gamma^{-\intercal}\right]}
$$

$$
\times \mathcal{P}_{ab,cd}\left(\frac{\partial}{\partial y}\right) e^{2\pi i\left(\frac{R}{i\sqrt{2}}y_{r\mu}(\gamma\Omega_2^{-1}\gamma^\intercal)^{rs}A_{si}^\intercal v^{\intercal i\mu} + y_{r\alpha}\widetilde{Q}_L{}^{r\alpha} + \frac{1}{4i}y_{r\alpha}(\gamma\Omega_2^{-1}\gamma^\intercal)^{rs}y_s{}^\alpha\right)},
$$

(D.1)

where a factor 2 comes from the center of order 2 of $GL(2, \mathbb{Z})$ acting on $\mathcal{H}_2$, $\gamma = \begin{pmatrix} p & q \\ r & s \end{pmatrix}$. The integral over positive definite matrices $\mathcal{P}_2$ splits into two Bessel-type integrals, using the

---

[34] Recall that $\mathrm{Dih}_4$ is the dihedral group of order 8 generated by the matrices $\begin{pmatrix} 1 & 0 \\ 0 & -1 \end{pmatrix}$ and $\begin{pmatrix} 0 & 1 \\ 1 & 0 \end{pmatrix}$, which stabilize $\begin{pmatrix} 0 & 1/2 \\ 1/2 & 0 \end{pmatrix}$.

projectors $\pi_1 = \begin{pmatrix} 1 & 0 \\ 0 & 0 \end{pmatrix}$, $\pi_2 = \begin{pmatrix} 0 & 0 \\ 0 & 1 \end{pmatrix}$

$$
\begin{aligned}
-\frac{R^4}{\pi} & \sum_{\widetilde{Q} \in \Lambda^{\oplus 2}_{p-2,q-2}} \sum_{\substack{A \in M_2(\mathbb{Z})/GL(2,\mathbb{Z}) \\ \gamma \in GL(2,\mathbb{Z})/\mathrm{Dih}_4}} e^{2\pi i a^{iI} A_{ij} \widetilde{Q}^j_I} c\Big(-\frac{(s\widetilde{Q}_1 - q\widetilde{Q}_2)^2}{2}\Big) c\Big(-\frac{(p\widetilde{Q}_2 - r\widetilde{Q}_1)^2}{2}\Big) \\
& \times \int_0^\infty \frac{\mathrm{d}\rho_2}{\rho_2} \rho_2^{\frac{q-6}{2}} e^{-\pi \mathrm{Tr}\left[\pi_1\left(\frac{R^2}{\rho_2 S_2} \gamma^\intercal A^\intercal \begin{pmatrix} 1 & S_1 \\ S_1 & |S|^2 \end{pmatrix} A\gamma + 2\rho_2 \gamma^{-1} \widetilde{Q} \cdot \widetilde{Q}^\intercal \gamma^{-\intercal}\right)\right]} \\
& \times \int_0^\infty \frac{\mathrm{d}\sigma_2}{\sigma_2} \sigma_2^{\frac{q-6}{2}} e^{-\pi \mathrm{Tr}\left[\pi_2\left(\frac{R^2}{\sigma_2 S_2} \gamma^\intercal A^\intercal \begin{pmatrix} 1 & S_1 \\ S_1 & |S|^2 \end{pmatrix} A\gamma + 2\sigma_2 \gamma^{-1} \widetilde{Q} \cdot \widetilde{Q}^\intercal \gamma^{-\intercal}\right)\right]} \\
& \times \mathcal{P}_{ab,cd}\Big(\frac{\partial}{\partial y}\Big) e^{2\pi i\Big(\frac{R}{\rho_2 i \sqrt{2}}(y_\mu \gamma \pi_1 \gamma^\intercal A^\intercal \nu^{\intercal \mu}) + y_\alpha \gamma \gamma^{-1} \widetilde{Q}_L{}^\alpha + \frac{1}{4i\rho_2}(y_\alpha \gamma \pi_1 \gamma^\intercal y^\alpha)\Big)} \\
& \times e^{2\pi i\Big(\frac{R}{\sigma_2 i \sqrt{2}}(y_\mu \gamma \pi_2 \gamma^\intercal A^\intercal \nu^{\intercal \mu}) + \frac{1}{4i\sigma_2}(y_\alpha \gamma \pi_2 \gamma^\intercal y^\alpha)\Big)} .
\end{aligned}
\tag{D.2}
$$

The matrices $\gamma \in GL(2,\mathbb{Z})/\mathrm{Dih}_4$ in the last two rows can be absorbed by a change of variable $(y_\mu, y_\alpha) \to (y_\mu \gamma^{-1}, y_\alpha \gamma^{-1})$. After relabelling the summation variable as $\binom{Q}{P} = A\binom{\widetilde{Q}_1}{\widetilde{Q}_2}$, one obtains a sum over all splittings $\Gamma = (Q,P) = \Gamma_1 + \Gamma_2$ in the lattice $\Lambda^{\oplus 2}_{p-2,q-2}$,

$$
\begin{aligned}
\sum_{\widetilde{Q}_i \in \Lambda^{\oplus 2}_{p-2,q-2}} & \sum_{\substack{A \in M_2(\mathbb{Z})/GL(2,\mathbb{Z}) \\ \gamma \in GL(2,\mathbb{Z})/\mathrm{Dih}_4}} e^{2\pi i a^{iI} A_{ij} \widetilde{Q}^j_I} f(A\gamma \pi_i \gamma^{-1} \widetilde{Q}) \, g((\pi_1 \gamma^{-1}\widetilde{Q})^2) g((\pi_2 \gamma^{-1}\widetilde{Q})^2) \\
& = \sum_{\Gamma \in \Lambda^{\oplus 2}_{p-2,q-2}} e^{2\pi i(a^1 \cdot Q + a^2 \cdot P)} \\
& \qquad \times \sum_{\substack{A \in M_2(\mathbb{Z})/GL(2,\mathbb{Z}) \\ \gamma \in GL(2,\mathbb{Z})/\mathrm{Dih}_4 \\ A^{-1}\Gamma \in \Lambda^{\oplus 2}_{p-2,q-2}}} f(A\gamma \pi_i (\gamma A)^{-1}\Gamma) \, g\Big(-\frac{(\pi_1 (\gamma A)^{-1}\Gamma)^2}{2}\Big) g\Big(-\frac{(\pi_2 (\gamma A)^{-1}\Gamma)^2}{2}\Big) \\
& = \sum_{\Gamma \in \Lambda^{\oplus 2}_{p-2,q-2}} e^{2\pi i(a^1 \cdot Q + a^2 \cdot P)} \sum_{\substack{B \in M_2(\mathbb{Z})/\mathrm{Stab}(\pi_i) \\ \Gamma_1, \Gamma_2 \in \Lambda_{p-2,q-2}}} f(\Gamma_1, \Gamma_2) \\
& \qquad \times \sum_{\Gamma_1/d_1 \in \Lambda_{p-2,q-2}} g\Big(-\frac{(B^{-1}\Gamma_1)^2}{2d_1^2}\Big) \sum_{\Gamma_2/d_2 \in \Lambda_{p-2,q-2}} g\Big(-\frac{(B^{-1}\Gamma_2)^2}{2d_2^2}\Big) .
\end{aligned}
\tag{D.3}
$$

where $\Gamma_i = B\pi_i B^{-1}\Gamma = (Q_i, P_i)$, such that $\Gamma_1 + \Gamma_2 = \Gamma$, and where $\mathrm{Stab}(\pi_i)$ is the stabilizer of $\pi_i = \begin{pmatrix} \delta_{1,i} & 0 \\ 0 & \delta_{2,i} \end{pmatrix}$ inside $M(2,\mathbb{Z})$. The rearrangement (D.3) holds for arbitrary functions $f(Q), g(x)$, in particular for the product of Bessel integrals and the measure factors $c(x)$ in (D.2). The singular contributions to the Fourier modes are thus

$$
\begin{aligned}
G^{(p,q),\,2\mathrm{Ab},\,\Gamma}_{\alpha\beta,\gamma\delta} = -\frac{R^4}{\pi} & \sum_{\substack{B \in M_2(\mathbb{Z})/\mathrm{Stab}(\pi_i) \\ B\pi_i B^{-1}\Gamma \in \Lambda^{\oplus 2}_{p-2,q-2}}} \bar{c}(\Gamma_1) \bar{c}(\Gamma_2) \sum_{l_1,l_2=0}^{2} \frac{P^{(l_1,l_2)}_{\alpha\beta,\gamma\delta}(\Gamma_1, \Gamma_2)}{R^{l_1+l_2}} \\
& \times \frac{K_{\frac{q-6}{2}-l_1}(2\pi R \mathcal{M}(\Gamma_1))}{\mathcal{M}(\Gamma_1)^{\frac{q-6}{2}-l_1}} \frac{K_{\frac{q-6}{2}-l_2}(2\pi R \mathcal{M}(\Gamma_2))}{\mathcal{M}(\Gamma_2)^{\frac{q-6}{2}-l_2}}
\end{aligned}
$$

$$G^{(p,q),\,2\mathrm{Ab},\,\Gamma}_{\alpha\beta,\gamma\upsilon} = -\frac{R^4}{\pi} \sum_{\substack{B \in M_2(\mathbb{Z})/\mathrm{Stab}(\pi_i) \\ B\pi_i B^{-1}\Gamma \in \Lambda^{\oplus 2}_{p-2,q-2}}} \bar{c}(\Gamma_1)\,\bar{c}(\Gamma_2) \sum_{l_1,l_2=0}^{1} \frac{P^{(l_1,l_2)}_{\alpha\beta,\gamma\upsilon}(\Gamma_1,\Gamma_2)}{\mathrm{i}\sqrt{2}R^{l_1+l_2}}$$

$$\times \frac{K_{\frac{q-6}{2}-l_1}(2\pi R\mathcal{M}(\Gamma_1))}{\mathcal{M}(\Gamma_1)^{\frac{q-6}{2}-l_1}} \frac{K_{\frac{q-6}{2}-l_2}(2\pi R\mathcal{M}(\Gamma_2))}{\mathcal{M}(\Gamma_2)^{\frac{q-6}{2}-l_2}}$$

$$\vdots \tag{D.4}$$

$$G^{(p,q),\,2\mathrm{Ab},\,\Gamma}_{\rho\sigma,\tau\upsilon} = -\frac{R^4}{\pi} \sum_{\substack{B \in M_2(\mathbb{Z})/\mathrm{Stab}(\pi_i) \\ B\pi_i B^{-1}\Gamma \in \Lambda^{\oplus 2}_{p-2,q-2}}} \bar{c}(\Gamma_1)\,\bar{c}(\Gamma_2) \sum_{l_1,l_2=0}^{1} \frac{P^{(l_1,l_2)}_{\rho\sigma,\tau\upsilon}(\Gamma_1,\Gamma_2)}{4R^{l_1+l_2}}$$

$$\times \frac{K_{\frac{q-6}{2}-l_1}(2\pi R\mathcal{M}(\Gamma_1))}{\mathcal{M}(\Gamma_1)^{\frac{q-6}{2}-l_1}} \frac{K_{\frac{q-6}{2}-l_2}(2\pi R\mathcal{M}(\Gamma_2))}{\mathcal{M}(\Gamma_2)^{\frac{q-6}{2}-l_2}},$$

where the measure $\bar{c}(\Gamma_i)$ is defined by

$$\bar{c}(\Gamma) = \sum_{\substack{d_i>0 \\ \Gamma/d_i \in \Lambda^{\oplus 2}_{p-2,q-2}}} c\left(-\frac{\gcd(Q^2,P^2,Q\cdot P)}{2d_i^2}\right)\left(\frac{d^2}{\gcd(Q^2,P^2,Q\cdot P)}\right)^{\frac{q-8}{2}}. \tag{D.5}$$

The factorized form of these singular contributions is indeed consistent with the differential equation (3.20), as discussed in §E.3.

## D.2 Measure factorization in CHL orbifolds

For CHL orbifolds, the contributions from the Dirac delta functions in (5.57) and (5.58) to the Fourier mode (5.56) can be computed similarly to the full rank case (D.4) by using the results of Appendix C. Here we explain the factorization of the measure for a general lattice $\Lambda_{p-2,q-2}$ of signature $(p-2,q-2)$, which we denote by $\Lambda$ for short. When the lattice is $N$-modular, as in the case of the magnetic lattice $\Lambda_m$ discussed in section C, one can rewrite the measure in a form manifestly invariant under Fricke electro-magnetic duality. However, this is not the case in generic signature. In this section we use the results of the previous section to write the 1/2-BPS charge measure factors coming from the different orbit terms in (5.64). By abuse of language we shall refer to the charges $(Q,P) \in \Lambda^* \oplus \Lambda$ components as electric and magnetic, although this terminology is only accurate when $q = 8$.

For the most generic lattice vectors, namely $(Q,P) \subseteq \Lambda^* \oplus \Lambda$, the only matrices $A$ which contribute belong either to the electric first orbit of the second set of splittings (C.36), or the magnetic second orbit (C.39) contribute. They both lead to the factorized measure

$$\bar{c}_k(\Gamma_1)\bar{c}_k(\Gamma_2) = \upsilon \sum_{\substack{d_1>0 \\ (Q_1,P_1)/d_1 \in \Lambda^* \oplus N\Lambda^*}} c_k\left(-\frac{\gcd(NQ_1^2,P_1^2,Q_1\cdot P_1)}{2d_1^2}\right)\left(\frac{d_1^2}{\gcd(NQ_1^2,P_1^2,Q_1\cdot P_1)}\right)^{\frac{q-8}{2}}$$

$$\times \sum_{\substack{d_2>0 \\ (Q_2,P_2)/d_2 \in \Lambda \oplus \Lambda}} c_k\left(-\frac{\gcd(Q_2^2,P_2^2,Q_2\cdot P_2)}{2d_2^2}\right)\left(\frac{d_2^2}{\gcd(Q_2^2,P_2^2,Q_2\cdot P_2)}\right)^{\frac{q-8}{2}}, \tag{D.6}$$

where $\Gamma_1$ is of electric type and $\Gamma_2$ of magnetic type. As explained in appendix C, this measure is consistent with splittings into pairs of 1/2-BPS charges of $(T,T)$ type.

For less generic vectors $(Q, P) \subsetneq \Lambda \oplus \Lambda$, the measure receives additional contributions from the first term of (5.56), as well as from the first magnetic orbit from the second set (C.37). Unlike the previous case $\Gamma_1$ can be either of electric or magnetic type, while $\Gamma_2$ is always of magnetic type. When $\Gamma_1$ is of electric type, the resulting measure is given by

$$
\bar{c}_k(\Gamma_1)\bar{c}_k(\Gamma_2) = \left[ \sum_{\substack{d_1 > 0 \\ (Q_1, P_1)/d_1 \in \Lambda \oplus N\Lambda}} c_k\left( -\frac{\gcd(Q_1^2, P_1^2, Q_1 \cdot P_1)}{2d_1^2} \right)\left( \frac{d_1^2}{\gcd(Q_1^2, P_1^2, Q_1 \cdot P_1)} \right)^{\frac{q-8}{2}} \right.
$$
$$
\left. + v \sum_{\substack{d_1 > 0 \\ (Q_1, P_1)/d_1 \in \Lambda^* \oplus N\Lambda^*}} c_k\left( -\frac{\gcd(NQ_1^2, P_1^2, Q_1 \cdot P_1)}{2d_1^2} \right)\left( \frac{d_1^2}{\gcd(NQ_1^2, P_1^2, Q_1 \cdot P_1)} \right)^{\frac{q-8}{2}} \right] \quad \text{(D.7)}
$$
$$
\times \sum_{\substack{d_2 > 0 \\ (Q_2, P_2)/d_2 \in \Lambda \oplus \Lambda}} c_k\left( -\frac{\gcd(Q_2^2, P_2^2, Q_2 \cdot P_2)}{2d_2^2} \right)\left( \frac{d_2^2}{\gcd(Q_2^2, P_2^2, Q_2 \cdot P_2)} \right)^{\frac{q-8}{2}},
$$

where only untwisted states can contribute in this case. This result is consistent with splittings of type $(U, T)$, as explained in Appendix C. When $\Gamma_1$ is of magnetic type, the measure is instead given by

$$
\bar{c}_k(\Gamma_1)\bar{c}_k(\Gamma_2) = \left[ \sum_{\substack{d_1 > 0 \\ (Q_1, P_1)/d_1 \in \Lambda \oplus \Lambda}} c_k\left( -\frac{\gcd(Q_1^2, P_1^2, Q_1 \cdot P_1)}{2d_1^2} \right)\left( \frac{d_1^2}{\gcd(Q_1^2, P_1^2, Q_1 \cdot P_1)} \right)^{\frac{q-8}{2}} \right.
$$
$$
\left. + v \sum_{\substack{d_1 > 0 \\ (Q_1, P_1)/d_1 \in N\Lambda^* \oplus N\Lambda^*}} c_k\left( -\frac{\gcd(NQ_1^2, P_1^2, Q_1 \cdot P_1)}{2Nd_1^2} \right)\left( \frac{Nd_1^2}{\gcd(NQ_1^2, P_1^2, Q_1 \cdot P_1)} \right)^{\frac{q-8}{2}} \right]
$$
$$
\times \sum_{\substack{d_2 > 0 \\ (Q_2, P_2)/d_2 \in \Lambda \oplus \Lambda}} c_k\left( -\frac{\gcd(Q_2^2, P_2^2, Q_2 \cdot P_2)}{2d_2^2} \right)\left( \frac{d_2^2}{\gcd(Q_2^2, P_2^2, Q_2 \cdot P_2)} \right)^{\frac{q-8}{2}},
$$
$$
\text{(D.8)}
$$

where both twisted and untwisted states can contribute, and where the former only get contributions form the second term in the bracket, while the latter get contributions from both, which is consistent with splittings into doublets of 1/2-BPS of $(\emptyset, UT)$ and $(\emptyset, TT)$ type.

For the vectors $(Q, P) \subsetneq \Lambda^* \oplus N\Lambda^*$, one must add to (D.6) the contribution from the last term of (5.56), *i.e.* both electric and magnetic orbits of the third set of contributions (C.26), (C.28). In this case, $\Gamma_1$ can only be of electric type, while $\Gamma_2$ can either be of electric or magnetic type. For electric $\Gamma_2$ one obtains

$$
\bar{c}_k(\Gamma_1)\bar{c}_k(\Gamma_2) = v \sum_{\substack{d_1 > 0 \\ (Q_1, P_1)/d_1 \in \Lambda^* \oplus N\Lambda^*}} c_k\left( -\frac{\gcd(NQ_1^2, P_1^2, Q_1 \cdot P_1)}{2d_1^2} \right)\left( \frac{d_1^2}{\gcd(NQ_1^2, P_1^2, Q_1 \cdot P_1)} \right)^{\frac{q-8}{2}}
$$
$$
\times \left[ \sum_{\substack{d_2 > 0 \\ (Q_2, P_2)/d_2 \in \Lambda \oplus N\Lambda}} c_k\left( -\frac{\gcd(Q_2^2, P_2^2, Q_2 \cdot P_2)}{2d_2^2} \right)\left( \frac{d_2^2}{\gcd(Q_2^2, P_2^2, Q_2 \cdot P_2)} \right)^{\frac{q-8}{2}} \right.
$$
$$
\left. + v \sum_{\substack{d_2 > 0 \\ (Q_2, P_2)/d_2 \in \Lambda^* \oplus N\Lambda^*}} c_k\left( -\frac{\gcd(NQ_2^2, P_2^2, Q_2 \cdot P_2)}{2d_2^2} \right)\left( \frac{d_2^2}{\gcd(NQ_2^2, P_2^2, Q_2 \cdot P_2)} \right)^{\frac{q-8}{2}} \right],
$$
$$
\text{(D.9)}
$$

where both twisted and untwisted states can contribute in this case, consistently with splittings of type $(TU,)$ and $(TT,)$. For magnetic $\Gamma_2$, one obtains

$$
\begin{aligned}
\bar{c}_k(\Gamma_1)\bar{c}_k(\Gamma_2) = v \sum_{\substack{d_1 > 0 \\ (Q_1,P_1)/d_1 \in \Lambda^* \oplus N\Lambda^*}} & c_k\Big(-\frac{\gcd(NQ_1^2, P_1^2, Q_1 \cdot P_1)}{2d_1^2}\Big)\Big(\frac{d_1^2}{\gcd(NQ_1^2, P_1^2, Q_1 \cdot P_1)}\Big)^{\frac{q-8}{2}} \\
\times \Bigg[ & \sum_{\substack{d_2 > 0 \\ (Q_2,P_2)/d_2 \in \Lambda \oplus \Lambda}} c_k\Big(-\frac{\gcd(Q_2^2, P_2^2, Q_2 \cdot P_2)}{2d_2^2}\Big)\Big(\frac{d_2^2}{\gcd(Q_2^2, P_2^2, Q_2 \cdot P_2)}\Big)^{\frac{q-8}{2}} \\
+ v & \sum_{\substack{d_2 > 0 \\ (Q_2,P_2)/d_2 \in N\Lambda^* \oplus N\Lambda^*}} c_k\Big(-\frac{\gcd(NQ_2^2, P_2^2, Q_2 \cdot P_2)}{2Nd_2^2}\Big)\Big(\frac{Nd_2^2}{\gcd(NQ_2^2, P_2^2, Q_2 \cdot P_2)}\Big)^{\frac{q-8}{2}} \Bigg],
\end{aligned}
$$

(D.10)

where only untwisted states can contribute, consistently with splittings of $(T, U)$ type. In both cases, the factors of $N$ come from the width of the integration domain $(\mathbb{R}/N\mathbb{Z})^3$.

Finally, for vectors $Q \in \Lambda$, $P \in N\Lambda^*$, one must add each contribution specific to the two last cases as well as the contribution from the second type of orbit of (5.51). Each 1/2-BPS state $\Gamma_1, \Gamma_2$ can be either electric or magnetic. When both of them are electric, we obtain

$$
\begin{aligned}
\bar{c}_k(\Gamma_1)\bar{c}_k(\Gamma_2) = \Bigg[ & \sum_{\substack{d_1 > 0 \\ (Q_1,P_1)/d_1 \in \Lambda \oplus N\Lambda}} c_k\Big(-\frac{\gcd(Q_1^2, P_1^2, Q_1 \cdot P_1)}{2d_1^2}\Big)\Big(\frac{d_1^2}{\gcd(Q_1^2, P_1^2, Q_1 \cdot P_1)}\Big)^{\frac{q-8}{2}} \\
+ v & \sum_{\substack{d_1 > 0 \\ (Q_1,P_1)/d_1 \in \Lambda^* \oplus N\Lambda^*}} c_k\Big(-\frac{\gcd(NQ_1^2, P_1^2, Q_1 \cdot P_1)}{2d_1^2}\Big)\Big(\frac{d_1^2}{\gcd(NQ_1^2, P_1^2, Q_1 \cdot P_1)}\Big)^{\frac{q-8}{2}} \Bigg] \\
\times \Bigg[ & \sum_{\substack{d_2 > 0 \\ (Q_2,P_2)/d_2 \in \Lambda \oplus N\Lambda}} c_k\Big(-\frac{\gcd(Q_2^2, P_2^2, Q_2 \cdot P_2)}{2d_2^2}\Big)\Big(\frac{d_2^2}{\gcd(Q_2^2, P_2^2, Q_2 \cdot P_2)}\Big)^{\frac{q-8}{2}} \\
+ v & \sum_{\substack{d_2 > 0 \\ (Q_2,P_2)/d_2 \in \Lambda^* \oplus N\Lambda^*}} c_k\Big(-\frac{\gcd(NQ_2^2, P_2^2, Q_2 \cdot P_2)}{2d_2^2}\Big)\Big(\frac{d_2^2}{\gcd(NQ_2^2, P_2^2, Q_2 \cdot P_2)}\Big)^{\frac{q-8}{2}} \Bigg],
\end{aligned}
$$

(D.11)

with constraints on the possible splittings, as explained in appendix C, selecting splittings of type $(TT,)$ only. When both states magnetic, one obtains

$$
\begin{aligned}
\bar{c}_k(\Gamma_1)\bar{c}_k(\Gamma_2) = \Bigg[ & \sum_{\substack{d_1 > 0 \\ (Q_1,P_1)/d_1 \in \Lambda \oplus \Lambda}} c_k\Big(-\frac{\gcd(Q_1^2, P_1^2, Q_1 \cdot P_1)}{2d_1^2}\Big)\Big(\frac{d_1^2}{\gcd(Q_1^2, P_1^2, Q_1 \cdot P_1)}\Big)^{\frac{q-8}{2}} \\
+ v & \sum_{\substack{d_1 > 0 \\ (Q_1,P_1)/d_1 \in N\Lambda^* \oplus N\Lambda^*}} c_k\Big(-\frac{\gcd(NQ_1^2, P_1^2, Q_1 \cdot P_1)}{2Nd_1^2}\Big)\Big(\frac{Nd_1^2}{\gcd(NQ_1^2, P_1^2, Q_1 \cdot P_1)}\Big)^{\frac{q-8}{2}} \Bigg] \\
\times \Bigg[ & \sum_{\substack{d_2 > 0 \\ (Q_2,P_2)/d_2 \in \Lambda \oplus \Lambda}} c_k\Big(-\frac{\gcd(Q_2^2, P_2^2, Q_2 \cdot P_2)}{2d_2^2}\Big)\Big(\frac{d_2^2}{\gcd(Q_2^2, P_2^2, Q_2 \cdot P_2)}\Big)^{\frac{q-8}{2}} \\
+ v & \sum_{\substack{d_2 > 0 \\ (Q_2,P_2)/d_2 \in N\Lambda^* \oplus N\Lambda^*}} c_k\Big(-\frac{\gcd(NQ_2^2, P_2^2, Q_2 \cdot P_2)}{2Nd_2^2}\Big)\Big(\frac{Nd_2^2}{\gcd(NQ_2^2, P_2^2, Q_2 \cdot P_2)}\Big)^{\frac{q-8}{2}} \Bigg],
\end{aligned}
$$

(D.12)

with again constraints on the possible splittings, selecting splittings of type $(\emptyset, TT)$ only. When one state, say $\Gamma_1$, is electric, and the other magnetic, one obtains

$$
\bar{c}_k(\Gamma_1)\bar{c}_k(\Gamma_2) = \left[ \sum_{\substack{d_1>0 \\ (Q_1,P_1)/d_1 \in \Lambda \oplus N\Lambda}} c_k\left(-\frac{\gcd(Q_1^2,P_1^2,Q_1\cdot P_1)}{2d_1^2}\right)\left(\frac{d_1^2}{\gcd(Q_1^2,P_1^2,Q_1\cdot P_1)}\right)^{\frac{q-8}{2}} \right.
$$
$$
\left. + \upsilon \sum_{\substack{d_1>0 \\ (Q_1,P_1)/d_1 \in \Lambda^* \oplus N\Lambda^*}} c_k\left(-\frac{\gcd(NQ_1^2,P_1^2,Q_1\cdot P_1)}{2d_1^2}\right)\left(\frac{d_1^2}{\gcd(NQ_1^2,P_1^2,Q_1\cdot P_1)}\right)^{\frac{q-8}{2}} \right]
$$
$$
\times \left[ \sum_{\substack{d_2>0 \\ (Q_2,P_2)/d_2 \in \Lambda \oplus \Lambda}} c_k\left(-\frac{\gcd(Q_2^2,P_2^2,Q_2\cdot P_2)}{2d_2^2}\right)\left(\frac{d_2^2}{\gcd(Q_2^2,P_2^2,Q_2\cdot P_2)}\right)^{\frac{q-8}{2}} \right.
$$
$$
\left. + \upsilon \sum_{\substack{d_2>0 \\ (Q_2,P_2)/d_2 \in N\Lambda^* \oplus N\Lambda^*}} c_k\left(-\frac{\gcd(NQ_2^2,P_2^2,Q_2\cdot P_2)}{2Nd_2^2}\right)\left(\frac{Nd_2^2}{\gcd(NQ_2^2,P_2^2,Q_2\cdot P_2)}\right)^{\frac{q-8}{2}} \right],
$$
(D.13)

where the constraints on the possible splitting here select $(U,U)$ only.

When the charge vectors $(Q,P)$ lies in an even finer sublattice, such as $\Lambda \oplus N\Lambda$, $N\Lambda^* \oplus N\Lambda^*$, and so on, the measure is still given by (D.13), but it includes less generic type of splittings like $(UU,)$ or $(\emptyset, UU)$, as explained in appendix C.

Thus, we have established that the delta function contributions to the Abelian Fourier coefficients factorize into the product $\bar{c}_k(\Gamma_1)\bar{c}_k(\Gamma_2)$ of the measures associated with each 1/2-BPS component for all splittings $\Gamma = \Gamma_1 + \Gamma_2$ of an arbitrary 1/4-BPS charge $\Gamma$ in CHL models with $N = 2, 3, 5, 7$. This factorization is required for consistency with the differential equation (3.20), as further discussed in §E.3.

# E  Consistency with differential constraints

In this section we analyze the consistency of the asymptotic expansion of the the two-loop modular integral $G_{ab,cd}$ near the degenerations $O(p,q) \to O(p-2,q-2)$ and degeneration $O(p,q) \to O(p-1,q-1)$ with the differential equation (3.3). In the first case we consider both the constant terms and generic rank-2 Abelian Fourier coefficients, and show consistency with the quadratic source term in (3.3). In the second case for brevity we restrict to the constant terms.

## E.1  Differential equation under the degeneration $O(p,q) \to O(p-2,q-2)$

Here we write explicitly the differential equation 3.3 in the variables relevant to the degeneration limit $O(p,q) \to O(p-2,q-2)$. Using the decomposition (5.3), and changing variable $R = e^{-\phi}$, the metric on the moduli space reads

$$
2P_{a\hat{b}}P^{a\hat{b}} = 4\mathrm{d}\phi^2 + 2P_{\mu\nu}P^{\mu\nu} + 2P_{\alpha\hat{\beta}}P^{\alpha\hat{\beta}} + e^{2\phi}M_{ij}g^{IJ}\mathrm{d}a_I^i\mathrm{d}a_J^j + e^{4\phi}\nabla\psi\nabla\psi,
$$
(E.1)

with

$$
\nabla\psi = \mathrm{d}\psi - \tfrac{1}{2}\varepsilon_{ij}a^i\cdot\mathrm{d}a^j,
$$
(E.2)

and the Maurer–Cartan coset component

$$
P = \begin{pmatrix}
\mathrm{d}\phi\,\delta_\mu{}^\nu - P_\mu{}^\nu & \frac{e^\phi}{\sqrt{2}} v_{i\mu}^{-1} p_L^{\beta I}\,\mathrm{d}a_I^i & -\frac{e^\phi}{\sqrt{2}} v_{i\mu}^{-1} p_R^{\hat\beta I}\,\mathrm{d}a_I^i & \frac{1}{2}e^{2\phi}\varepsilon_\mu{}^{\hat\nu}\nabla\psi \\
\frac{e^\phi}{\sqrt{2}} v_i^{-1\,\nu} p_{L\alpha}{}^I\,\mathrm{d}a_I^i & 0 & P_\alpha{}^{\hat\beta} & \frac{e^\phi}{\sqrt{2}} v_i^{-1\,\hat\nu} p_{L\alpha}{}^I\,\mathrm{d}a_I^i \\
-\frac{e^\phi}{\sqrt{2}} v_i^{-1\,\nu} p_{R\hat\alpha}{}^I\,\mathrm{d}a_I^i & P^\beta{}_{\hat\alpha} & 0 & \frac{e^\phi}{\sqrt{2}} v_i^{-1\,\hat\nu} p_{R\hat\alpha}{}^I\,\mathrm{d}a_I^i \\
\frac{1}{2}e^{2\phi}\varepsilon^\nu{}_{\hat\mu}\nabla\psi & \frac{e^\phi}{\sqrt{2}} v_{i\hat\mu}^{-1} p_L^{\beta I}\,\mathrm{d}a_I^i & \frac{e^\phi}{\sqrt{2}} v_{i\hat\mu}^{-1} p_R^{\hat\beta I}\,\mathrm{d}a_I^i & -\mathrm{d}\phi\,\delta_{\hat\mu}{}^{\hat\nu} + P_{\hat\mu}{}^{\hat\nu}
\end{pmatrix}. \tag{E.3}
$$

Beware that in this section we use the symbols $p_L$ and $p_R$ for the $G_{p-2,q-2} = O(p-2,q-2)/[O(p-2)\times O(q-2)]$ projection $p_{LaI}Q^I$, and not for the $G_{p,q} = O(p,q)/[O(p)\times O(q)]$ projection $p_{La\mathcal{I}}Q^{\mathcal{I}}$ as in the body of the paper. We use Greek letters of the beginning of the alphabet, *i.e.* $\{\alpha,\beta,\gamma,\delta,\epsilon,\eta,\theta\}$, to denote local indices along $G_{p-2,q-2}$, and Greek letters of the middle of the alphabet, *i.e.* $\{\kappa,\lambda,\mu,\nu,\rho,\sigma,\tau\}$, to denote indices along $SO(2)\backslash SL(2,\mathbb{R})$.

The covariant derivative of a vector $Z_a$ in the tangent frame must obey the usual equation

$$
\mathrm{d}Z_a = 2P^{b\hat c}\partial_{b\hat c}Z_a = 2P^{b\hat c}\left(D_{b\hat c}Z_a - B_{b\hat c a}{}^d Z_d\right), \tag{E.4}
$$

allowing us to write down its action, for any vector $Z_a = (Z_\sigma, Z_\gamma)$

$$
\begin{aligned}
D_{\mu\hat\nu}Z_a &= \left(\tfrac{1}{4}\delta_{\mu\hat\nu}\partial_\phi - \mathcal{D}_{\mu\hat\nu} + \tfrac{1}{2}e^{-2\phi}\varepsilon_{\mu\hat\nu}\partial_\psi\right)Z_a + \tfrac{1}{2}(\delta_{\sigma[\mu}\delta_{\hat\nu]}^\rho Z_\rho, 0), \\
D_{\alpha\hat\nu}Z_a &= \tfrac{1}{\sqrt{2}}e^{-\phi}v_{\hat\nu}{}^i p_{L\alpha}{}^I\left(\frac{\partial}{\partial a^{iI}} - \tfrac{1}{2}\varepsilon_{ij}a_I^j\partial_\psi\right)Z_a + \tfrac{1}{2}(-\delta_{\hat\nu\sigma}Z_\alpha, \delta_{\alpha\gamma}\delta_{\hat\nu}^\nu Z_\nu), \\
D_{\mu\hat\alpha}Z_a &= \tfrac{1}{\sqrt{2}}e^{-\phi}v_\mu{}^i p_{R\hat\alpha}{}^I\left(\frac{\partial}{\partial a^{iI}} - \tfrac{1}{2}\varepsilon_{ij}a_I^j\partial_\psi\right)Z_a,
\end{aligned} \tag{E.5}
$$

and on any vector $Z_{\hat a} = (Z_{\hat\alpha}, Z_{\hat\sigma})$ as

$$
\begin{aligned}
D_{\mu\hat\nu}Z_{\hat a} &= \left(\tfrac{1}{4}\delta_{\mu\hat\nu}\partial_\phi - \mathcal{D}_{\mu\hat\nu} + \tfrac{1}{2}e^{-2\phi}\varepsilon_{\mu\hat\nu}\partial_\psi\right)Z_{\hat a} + \tfrac{1}{2}(0, -\delta_{\hat\sigma[\mu}\delta_{\hat\nu]}^{\hat\rho}Z_{\hat\rho}), \\
D_{\alpha\hat\nu}Z_{\hat a} &= \tfrac{1}{\sqrt{2}}e^{-\phi}v_{\hat\nu}{}^i p_{L\alpha}{}^I\left(\frac{\partial}{\partial a^{iI}} - \tfrac{1}{2}\varepsilon_{ij}a_I^j\partial_\psi\right)Z_{\hat a}, \\
D_{\mu\hat\alpha}Z_{\hat a} &= \tfrac{1}{\sqrt{2}}e^{-\phi}v_\mu{}^i p_{R\hat\alpha}{}^I\left(\frac{\partial}{\partial a^{iI}} - \tfrac{1}{2}\varepsilon_{ij}a_I^j\partial_\psi\right)Z_{\hat a} + \tfrac{1}{2}(\delta_{\hat\alpha\hat\alpha}\delta_\mu^{\hat\mu}Z_{\hat\mu}, -\delta_{\mu\hat\sigma}Z_{\hat\alpha}),
\end{aligned} \tag{E.6}
$$

where $v_\mu{}^i \in SO(2)\backslash SL(2,\mathbb{R})$ such that

$$
\mathcal{D}_{\mu\nu}v_\rho{}^i = \tfrac{1}{2}\delta_{\rho(\mu}v_{\nu)}{}^i - \tfrac{1}{4}\delta_{\mu\nu}v_\rho{}^i, \tag{E.7}
$$

and finally, the operator $D_{\alpha\hat\beta} = \mathcal{D}_{\alpha\hat\beta}$ the differential operator on the Grassmanian $O(p-2,q-2)$, which acts on the projectors $p_{L\gamma}{}^I$, $p_{R\hat\alpha}{}^I$ as

$$
\mathcal{D}_{\alpha\hat\beta}p_{L\gamma}{}^I = \tfrac{1}{2}\delta_{\alpha\gamma}p_{L\hat\beta}{}^I, \qquad \mathcal{D}_{\alpha\hat\beta}p_{R\hat\alpha}{}^I = \tfrac{1}{2}\delta_{\hat\beta\hat\alpha}p_{L\alpha}{}^I. \tag{E.8}
$$

In this decomposition, the tensor $G_{ab,cd}$ admits six independent components

$$
G_{\mu\nu,\sigma\rho} = \tfrac{3}{4}\delta_{\langle\mu\nu}\delta_{\sigma\rho\rangle}G_\lambda{}^\lambda{}_{,\kappa}{}^\kappa, \quad G_{\mu\nu,\sigma\delta} = \delta_{\mu\nu}G_{\sigma\delta,\lambda}{}^\lambda - \delta_{\sigma(\mu}G_{\nu)\delta,\lambda}{}^\lambda, \quad G_{\mu\nu,\gamma\delta},
$$

$$
G_{\mu\beta,\nu\delta} = G_{\beta[\mu,\nu]\delta} - \tfrac{1}{2}G_{\mu\nu,\beta\delta}, \quad G_{\mu\beta,\gamma\delta}, \quad G_{\alpha\beta,\gamma\delta}, \tag{E.9}
$$

but for simplicity we shall only consider the components $G_{\mu\nu,\sigma\rho}$, $G_{\mu\nu,\gamma\delta}$ and $G_{\alpha\beta,\gamma\delta}$ that admit a non-trivial constant term. The differential operator $D_{(\mu}{}^{\hat c}D_{\nu)\hat c}G_{ab,cd}$ acts diagonally on the

various components of fixed number of indices along the Grassmanian, so it is consistant to only consider the components with an even number of indices along the sub-Grassmaniann in the differential equation (3.20). Using the Fourier decompositions

$$
\begin{aligned}
G_{ab,cd} &= \sum_{\Gamma \in \Lambda^* \oplus \Lambda} G^{\Gamma}_{ab,cd} e^{2\pi i(\Gamma, a)} + \sum_{n \neq 0} G^{\mathrm{TN}\, n}_{ab,cd} e^{2\pi i n \psi} , \\
F_{abcd} &= \sum_{\Gamma \in \Lambda^* \oplus \Lambda} F^{\Gamma}_{abcd} e^{2\pi i(\Gamma, a)} + \sum_{n \neq 0} F^{\mathrm{TN}\, n}_{abcd} e^{2\pi i n \psi} ,
\end{aligned} \tag{E.10}
$$

one obtains from (3.20)

$$
\begin{aligned}
&\left( 2\mathcal{D}_{(\mu}{}^{\tau}\mathcal{D}_{\nu)\tau} - (\partial_\phi + q - 2)\mathcal{D}_{\mu\nu} + \tfrac{1}{8}(\partial_\phi + 8)(\partial_\phi + 2q - 10)\delta_{\mu\nu} - 4\pi e^{-2\phi}\Gamma_{R\mu}\cdot\Gamma_{R\nu} \right) G^{\Gamma}_{\sigma\rho,\kappa\lambda} \\
=\ & -\frac{3\pi}{4}\delta_{\langle\sigma\rho,}\delta_{\kappa\lambda\rangle} \sum_{\Gamma_1 \in \Lambda^* \oplus \Lambda} \left( F^{\Gamma_1}{}_{\kappa d(\mu}{}^{\kappa}F^{\Gamma-\Gamma_1\, \lambda d}_{\nu)\lambda} - F^{\Gamma_1}{}_{\kappa d(\mu}{}^{\lambda}F^{\Gamma-\Gamma_1\, \kappa d}_{\nu)\lambda} \right) - 3\pi F^{\Gamma}_{\mu\nu,\sigma\rho,\kappa\lambda} ,
\end{aligned} \tag{E.11}
$$

$$
\begin{aligned}
&\left( 2\mathcal{D}_{(\mu}{}^{\tau}\mathcal{D}_{\nu)\tau} - (\partial_\phi + q - 2)\mathcal{D}_{\mu\nu} + \tfrac{1}{8}(\partial_\phi + 6)(\partial_\phi + 2q - 8)\delta_{\mu\nu} - 4\pi e^{-2\phi}\Gamma_{R\mu}\cdot\Gamma_{R\nu} \right) G^{\Gamma}_{\sigma\rho,\gamma\delta} \\
=\ & \frac{1}{2}\delta_{\mu\nu}G^{\Gamma}_{\sigma\rho,\gamma\delta} + \frac{8-q}{2}\delta_{\sigma\rho}G^{\Gamma}_{\mu\nu\gamma\delta} + \frac{6-q}{2}\delta_{\sigma(\mu}\delta_{\nu)\rho}G^{\Gamma}_{\alpha\beta,\lambda}{}^{\lambda} + \frac{2q-13}{2}\delta_{\mu\nu}\delta_{\sigma\rho}G^{\Gamma}_{\alpha\beta,\lambda}{}^{\lambda} \\
& + \mathcal{D}_{(\mu}{}^{\lambda}\delta_{\nu)(\sigma}G^{\Gamma}_{\rho)\lambda,\alpha\beta} - \mathcal{D}_{\sigma)(\mu}G^{\Gamma}_{\nu)(\rho|,\alpha\beta} + \delta_{\alpha\beta}G^{\Gamma}_{\mu\nu,\sigma\rho} \\
& - 2\pi \sum_{\Gamma_1 \in \Lambda^* \oplus \Lambda} \left( F^{\Gamma_1}{}_{\sigma\rho d(\mu}{}^{\kappa}F^{\Gamma-\Gamma_1\, d}_{\nu)\gamma\delta} - F^{\Gamma_1}{}_{\sigma)\gamma d(\mu}F^{\Gamma-\Gamma_1\, d}_{\nu)\delta(\rho} \right) - 3\pi F^{\Gamma}_{\mu\nu,\sigma\rho,\gamma\delta} ,
\end{aligned} \tag{E.12}
$$

and

$$
\begin{aligned}
&\left( 2\mathcal{D}_{(\mu}{}^{\tau}\mathcal{D}_{\nu)\tau} - (\partial_\phi + q - 2)\mathcal{D}_{\mu\nu} + \tfrac{1}{8}(\partial_\phi + 4)(\partial_\phi + 2q - 6)\delta_{\mu\nu} - 4\pi e^{-2\phi}\Gamma_{R\mu}\cdot\Gamma_{R\nu} \right) G^{\Gamma}_{\alpha\beta,\gamma\delta} \\
=\ & 3\delta_{\langle\alpha\beta,}G^{\Gamma}_{\gamma\delta\rangle,\mu\nu} - 3\pi \sum_{\Gamma_1 \in \Lambda^* \oplus \Lambda} F^{\Gamma_1}{}_{\mu)d\langle\alpha\beta,}F^{\Gamma-\Gamma_1\, d}_{\gamma\delta\rangle(\nu} - 3\pi F^{\Gamma}_{\mu\nu,\alpha\beta,\gamma\delta} ,
\end{aligned} \tag{E.13}
$$

where the additional term of order $\mathcal{O}(e^{-R^2})$ comes from the Abelian Fourier coefficients of the quadratic source in $F_{abcd}$ involving nonzero Taub-NUT charge,

$$
\sum_{n \neq 0} F^{\mathrm{TN}\, n}_{eg\langle ab,} F^{\mathrm{TN}\, -n\, g}_{cd\rangle f} = \sum_{\Gamma \in \Lambda^* \oplus \Lambda} F^{\Gamma}_{ef,ab,cd} e^{2\pi i(\Gamma, a)} . \tag{E.14}
$$

It is a non-trivial task to compute these Fourier coefficients from the explicit non-Abelian Fourier coefficients of the tensor $F_{abcd}$, which we shall attempt to carry out in this paper.

Introducing for brevity the vector $\vec{G}_\Gamma$

$$
\vec{G}_\Gamma = (G^{\Gamma}_{\rho\sigma,\tau\upsilon}, G^{\Gamma}_{\rho\sigma,\gamma\delta}, G^{\Gamma}_{\alpha\beta,\gamma\delta}), \tag{E.15}
$$

we find that the differential operator with two indices along the sub-Grassmaniann acts on $\vec{G}_\Gamma$ according to

$$
\begin{aligned}
4D_{(\eta}{}^{\hat{\eta}}D_{\theta)\hat{\eta}}\vec{G}_\Gamma =\ & (4\mathcal{D}_{(\eta}{}^{\hat{\eta}}\mathcal{D}_{\theta)\hat{\eta}} + \delta_{\eta\theta}\partial_\phi - 8\pi^2 e^{-2\phi}\Gamma_{L\eta}{}^{\kappa}\Gamma_{L\theta\kappa})\vec{G}_\Gamma \\
& + 8i\pi\sqrt{2}e^{-\phi}\Gamma_{L(\eta|}{}^{\kappa} \begin{pmatrix} -\delta_{\kappa(\rho}G^{\Gamma}_{\sigma)|\theta),\tau\upsilon} - \delta_{\kappa(\tau}G^{\Gamma}_{\upsilon)|\theta),\rho\sigma} \\ -\delta_{\kappa(\rho}G^{\Gamma}_{\sigma)|\theta),\gamma\delta} + \delta_{\theta)(\gamma}G^{\Gamma}_{\delta)\kappa,\rho\sigma} \\ \delta_{\theta)(\alpha}G^{\Gamma}_{\beta)\kappa,\gamma\delta} + \delta_{\theta)(\gamma}G^{\Gamma}_{\delta)\kappa,\alpha\beta} \end{pmatrix} , \\
& + \begin{pmatrix} 6\delta_{\langle\rho\sigma,}G^{\Gamma}_{\tau\upsilon),\gamma\delta} - 4\delta_{\eta\theta}G^{\Gamma}_{\rho\sigma,\tau\upsilon} \\ 2\delta_{\rho\sigma}G^{\Gamma}_{\eta\theta,\gamma\delta} - 2\delta_{\eta\theta}G^{\Gamma}_{\rho\sigma,\gamma\delta} + 2\delta_{\eta(\gamma}\delta_{\delta)\theta}G^{\Gamma}_{\rho\sigma,\kappa}{}^{\kappa} \\ 6\delta_{\eta\langle(\alpha}\delta_{\beta),|\theta|}G^{\Gamma}_{\gamma\delta),\kappa}{}^{\kappa} - 8\delta_{(\eta|\langle(\alpha}G^{\Gamma}_{\beta),|\theta),|\gamma\delta\rangle} \end{pmatrix} ,
\end{aligned} \tag{E.16}
$$

where the term linear in $\Gamma$ involves the components of $G^\Gamma$ with an odd number of indices along the sub-Grassmaniann. Using this action, we find the differential equation obeyed by the components with two indices along the sub-Grassmaniann $G_{p-2,q-2}$,

$$\left(2\mathcal{D}_{(\eta}{}^{\hat{a}}\mathcal{D}_{\alpha)\hat{\zeta}} + \tfrac{1}{2}\delta_{\eta\theta}\partial_\phi - 4\pi^2 e^{-2\phi}\Gamma_{L\eta}{}^\kappa\Gamma_{L\theta\kappa}\right)\vec{G}_\Gamma$$

$$+ 4i\pi\sqrt{2}e^{-\phi}\Gamma_{L(\eta|}{}^\kappa \begin{pmatrix} -\delta_{\kappa(\rho}G^\Gamma_{\sigma)|\theta),\tau\upsilon} - \delta_{\kappa(\tau}G^\Gamma_{\upsilon)|\theta),\rho\sigma} \\ -\delta_{\kappa(\rho}G^\Gamma_{\sigma)|\theta),\gamma\delta} + \delta_{\theta)(\gamma}G^\Gamma_{\delta)\kappa,\rho\sigma} \\ \delta_{|\theta)(\alpha}G^\Gamma_{\beta)\kappa,\gamma\delta} + \delta_{\theta)(\gamma}G^\Gamma_{\delta)\kappa,\alpha\beta} \end{pmatrix}$$

$$= \begin{pmatrix} (5-q)\delta_{\eta\theta}G^\Gamma_{\rho\sigma,\tau\upsilon} \\ (4-q)\delta_{\eta\theta}G^\Gamma_{\rho\sigma,\gamma\delta} + (6-q)\delta_{|\eta)(\gamma}G^\Gamma_{\delta)(\theta|,\rho\sigma} + \delta_{\gamma\delta}G^\Gamma_{\eta\theta,\rho\sigma} - \delta_{\theta(\gamma}\delta_{\delta)\eta}G^\Gamma_{\rho\sigma,\lambda}{}^\lambda \\ (3-q)\delta_{\eta\theta}G^\Gamma_{\alpha\beta,\gamma\delta} + 2(8-q)\delta_{|\eta)((\alpha}G^\Gamma_{\beta),(\theta|,|\gamma\delta)} + 3\delta_{\langle\alpha\beta,}G^\Gamma_{\gamma\delta\rangle,\eta\theta} - 3\delta_{\eta((\alpha}\delta_{\beta),|\theta|}G^\Gamma_{\gamma\delta),\kappa}{}^\kappa \end{pmatrix}$$

$$- \pi \sum_{\Gamma_1 \in \Lambda^* \oplus \Lambda} \begin{pmatrix} 3F^{\Gamma_1}_{\eta d\langle\rho\sigma,}F^{\Gamma-\Gamma_1 d}_{\tau\upsilon\rangle\theta} \\ 2F^{\Gamma_1}_{\rho\sigma d(\eta}{}^\kappa F^{\Gamma-\Gamma_1 d}_{\theta)\gamma\delta} - 2F^{\Gamma_1}_{\sigma)\gamma d(\eta}F^{\Gamma-\Gamma_1\ d}_{\theta)\delta(\rho} \\ 3F^{\Gamma_1}_{\eta d\langle\alpha\beta,}F^{\Gamma-\Gamma_1 d}_{\gamma\delta\rangle\theta} \end{pmatrix} - 3\pi\vec{F}_{\eta\theta\,\Gamma} \,. \quad \text{(E.17)}$$

## E.2 Zero mode equations

In this subsection we analyze the consistency of the differential equations (E.11), (E.12), (E.13), (E.17) with the results in §5 for the constant term $G^0_{ab,cd}$. As mentioned earlier, the unfolding method fails to capture exponentially suppressed corrections to the constant term, which are sourced by the quadratic terms $\sum_{\Gamma_1 \neq 0} F^{\Gamma_1}F^{-\Gamma_1}$ and $F^0_{ef,ab,cd}$ defined in (E.14) on the right-hand side of the differential equation, and can be ascribed to instanton anti-instanton configurations. These terms can in principle be computed by solving the differential equation. Here we concentrate on the perturbative part of $G^0_{ab,cd}$, which is sourced by the square of the perturbative part of $F_{abcd}$. The latter is given by [22, (5.29)]

$$F^0_{\mu\nu\rho\sigma} = 4e^{(6-q)\phi}\Big(\mathcal{D}_{(\mu\nu}\mathcal{D}_{\rho\sigma)} + \tfrac{q-10}{2}\delta_{(\mu\nu}\mathcal{D}_{\sigma\rho)} + \tfrac{(8-q)(12-q)}{16}\delta_{(\mu\nu}\delta_{\rho\sigma)}\Big)\mathcal{E}(S)\,,$$

$$F^0_{\mu\nu\gamma\delta} = e^{(6-q)\phi}\delta_{\gamma\delta}\Big(\tfrac{8-q}{2}\delta_{\mu\nu} - 2\mathcal{D}_{\mu\nu}\Big)\mathcal{E}(S)\,,$$

$$F^0_{\alpha\beta\gamma\delta} = e^{-2\phi}\mathcal{F}_{\alpha\beta\gamma\delta}(\varphi) + 3e^{(6-q)\phi}\delta_{(\alpha\beta}\delta_{\gamma\delta)}\mathcal{E}(S)\,, \quad \text{(E.18)}$$

where

$$\mathcal{E}(S) = \frac{3}{(N+1)\pi^2}\Big(\mathcal{E}^\star(\tfrac{8-q}{2},S) + \upsilon N^{\frac{8-q}{2}}\mathcal{E}^\star(\tfrac{8-q}{2},NS)\Big) \quad \text{(E.19)}$$

is a specific solution of the Laplace equation

$$2\mathcal{D}_{\rho\sigma}\mathcal{D}^{\rho\sigma}\mathcal{E}(S) = \frac{1}{2}(D_{-2}\bar{D}_0 + \bar{D}_2 D_0)\mathcal{E}(S) = \frac{(8-q)(6-q)}{4}\mathcal{E}(S)\,. \quad \text{(E.20)}$$

It is then straightforward to find a particular solution to Eq. (E.17)

$$G^0_{\mu\nu,\rho\sigma} = -3\pi e^{2(6-q)\phi}\delta_{\langle\mu\nu,}\delta_{\rho\sigma\rangle}\Big(\big(\tfrac{8-q}{2}\big)^2\mathcal{E}(S)^2 - 2\mathcal{D}_{\kappa\lambda}\mathcal{E}(S)\mathcal{D}^{\kappa\lambda}\mathcal{E}(S)\Big)\,,$$

$$G^0_{\mu\nu,\gamma\delta} = -\tfrac{\pi}{6}e^{(4-q)\phi}\big(\tfrac{8-q}{2}\delta_{\mu\nu} - 2\mathcal{D}_{\mu\nu}\big)\mathcal{E}(S)\mathcal{G}_{\gamma\delta}(\varphi) - 2\pi e^{2(6-q)\phi}\delta_{\gamma\delta}\mathcal{E}(S)\big(\tfrac{8-q}{2}\delta_{\mu\nu} - 2\mathcal{D}_{\mu\nu}\big)\mathcal{E}(S)\,,$$

$$G^0_{\alpha\beta,\gamma\delta} = e^{-4\phi}\mathcal{G}_{\alpha\beta,\gamma\delta}(\varphi) - \tfrac{\pi}{2}e^{(4-q)\phi}\mathcal{E}(S)\delta_{\langle\alpha\beta,}\mathcal{G}_{\gamma\delta\rangle}(\varphi) - 3\pi e^{2(6-q)\phi}\delta_{\langle\alpha\beta,}\delta_{\gamma\delta\rangle}\mathcal{E}(S)^2\,, \quad \text{(E.21)}$$

with $\mathcal{G}_{\alpha\beta,\gamma\delta}(\varphi)$ solution to an equation analogue to (3.20) with source term quadratic in $\mathcal{F}_{\alpha\beta\gamma\delta}(\varphi)$, and $\mathcal{G}_{\alpha\beta}(\varphi)$ solution to the equation on the sub-Grassmaniann $G_{p-2,q-2}$

$$2\mathcal{D}_{(\gamma}{}^{\hat{a}}\mathcal{D}_{\delta)\hat{a}}\mathcal{G}_{\alpha\beta} = \tfrac{4-q}{2}\delta_{\gamma\delta}\mathcal{G}_{\alpha\beta} + (6-q)\delta_{\gamma)(\alpha}\mathcal{G}_{\beta)(\delta} + \delta_{\alpha\beta}\mathcal{G}_{\gamma\delta} + 12\mathcal{F}_{\alpha\beta\gamma\delta}\,. \quad \text{(E.22)}$$

One can then check that $G^0_{ab,cd}$ is also a solution to (E.11), (E.12) and (E.13), using the identity

$$F^0_{\kappa d(\mu}{}^\kappa F^0_{\nu)\lambda}{}^{\lambda d} - F^0_{\kappa d(\mu}{}^\lambda F^0_{\nu)\lambda}{}^{\kappa d} = 2(8-q)^2\left(\left(\frac{8-q}{2}\right)^2+1\right)\delta_{\mu\nu}\mathcal{E}^2 - (8-q)^2(10-q)\mathcal{E}\mathcal{D}_{\mu\nu}\mathcal{E}$$
$$+ 8(10-q)\mathcal{D}_{\mu\nu}\mathcal{D}_{\rho\sigma}\mathcal{E}\mathcal{D}^{\rho\sigma}\mathcal{E} - 16\mathcal{D}_\mu{}^\lambda\mathcal{D}_{\rho\sigma}\mathcal{E}\mathcal{D}_{\nu\lambda}\mathcal{D}^{\rho\sigma}\mathcal{E} \quad \text{(E.23)}$$

and the fact that for any two symmetric tensors $X_{\mu\nu}$ and $Y_{\mu\nu}$, one has

$$X_{\langle\mu\nu}Y_{\rho\sigma\rangle} = \tfrac{1}{2}\delta_{\langle\mu\nu},\delta_{\rho\sigma\rangle}(X_\lambda{}^\lambda Y_\kappa{}^\kappa - X^{\kappa\lambda}Y_{\kappa\lambda})\,. \quad \text{(E.24)}$$

The most general solution is obtained by adding a solution of the homogeneous equation without source term, given by

$$\tilde{G}^0_{\mu\nu,\rho\sigma} = \frac{(6-q)(7-q)}{2}c\,e^{2(5-q)\phi}\delta_{\langle\mu\nu},\delta_{\rho\sigma\rangle}\,,$$
$$\tilde{G}^0_{\mu\nu,\gamma\delta} = -\frac{\pi}{6}e^{(4-q)\phi}\left(\frac{8-q}{2}\delta_{\mu\nu} - 2\mathcal{D}_{\mu\nu}\right)\tilde{\mathcal{E}}(S)\tilde{\mathcal{G}}_{\gamma\delta}(\varphi) + \frac{7-q}{3}c\,e^{2(5-q)\phi}\delta_{\mu\nu}\delta_{\gamma\delta}\,,$$
$$\tilde{G}^0_{\alpha\beta,\gamma\delta} = e^{-4\phi}\tilde{\mathcal{G}}_{\alpha\beta,\gamma\delta}(\varphi) - \frac{\pi}{2}e^{(4-q)\phi}\tilde{\mathcal{E}}(S)\delta_{\langle\alpha\beta},\tilde{\mathcal{G}}_{\gamma\delta\rangle}(\varphi) + c\,e^{2(5-q)\phi}\delta_{\langle\alpha\beta},\delta_{\gamma\delta\rangle}\,, \quad \text{(E.25)}$$

with $c$ a numerical constant, $\tilde{\mathcal{G}}_{\alpha\beta,\gamma\delta}(\varphi)$ a solution to the homogeneous equation (3.17) on the sub-Grassmanian, $\tilde{\mathcal{E}}$ a solution to (E.20) and $\tilde{\mathcal{G}}_{\alpha\beta}(\varphi)$ solution to the homogeneous equation (3.34) on the sub-Grassmanian. The explicit results (5.44), (5.60) for the constant term $G^0_{ab,cd}$ obtained by unfolding method for generic values of $q$ indeed lie in this class, upon setting

$$\tilde{\mathcal{E}}(S) = \frac{3}{(N-1)\pi^2}\left(-\mathcal{E}^\star(\tfrac{8-q}{2},S) + \upsilon N^{\frac{8-q}{2}}\mathcal{E}^\star(\tfrac{8-q}{2},NS)\right),$$
$$\mathcal{G}_{\alpha\beta}(\varphi) = \tfrac{1}{2}\left(G^{(p-2,q-2)}_{\alpha\beta}(\varphi) + {}^\varsigma G^{(p-2,q-2)}_{\alpha\beta}(\varphi)\right),$$
$$\tilde{\mathcal{G}}_{\alpha\beta}(\varphi) = \tfrac{1}{2}\left(G^{(p-2,q-2)}_{\alpha\beta}(\varphi) - {}^\varsigma G^{(p-2,q-2)}_{\alpha\beta}(\varphi)\right),$$
$$c = \frac{18}{\pi^2}\xi(7-q)\xi(6-q)\frac{(N-\upsilon)(1-\upsilon N^{q-7})}{N^2-1}\,. \quad \text{(E.26)}$$

For special values of $q$ one must take into account additional source terms due to logarithmic divergences. For example for $q = 8$, one has instead

$$F^0_{\mu\nu\rho\sigma} = e^{-2\phi}\left(4(\mathcal{D}_{(\mu\nu}\mathcal{D}_{\rho\sigma)} - \delta_{(\mu\nu}\mathcal{D}_{\sigma\rho)})\hat{\mathcal{E}}(S) - 2\kappa\delta_{(\mu\nu}\delta_{\rho\sigma)}\right),$$
$$F^0_{\mu\nu\gamma\delta} = -e^{-2\phi}\delta_{\gamma\delta}\left(\kappa\delta_{\mu\nu} + 2\mathcal{D}_{\mu\nu}\hat{\mathcal{E}}(S)\right),$$
$$F^0_{\alpha\beta\gamma\delta} = e^{-2\phi}\left(\hat{\mathcal{F}}_{\alpha\beta\gamma\delta}(\varphi) + 3\delta_{(\alpha\beta}\delta_{\gamma\delta)}(\hat{\mathcal{E}}(S) - 2\kappa\phi)\right), \quad \text{(E.27)}$$

where $\hat{\mathcal{E}}(S) = \frac{1}{2\pi(N+1)}\left(\hat{\mathcal{E}}_1(S) + \hat{\mathcal{E}}_1(NS)\right)$ satisfies a Poisson equation with a constant source term,

$$2\mathcal{D}_{\rho\sigma}\mathcal{D}^{\rho\sigma}\hat{\mathcal{E}}(S) = \tfrac{1}{2}(D_{-2}\bar{D}_0 + \bar{D}_2 D_0)\hat{\mathcal{E}}(S) = \kappa\,. \quad \text{(E.28)}$$

One finds the particular solution to E. (E.17),

$$G^0_{\mu\nu,\rho\sigma} = -3\pi e^{-4\phi}\delta_{\langle\mu\nu},\delta_{\rho\sigma\rangle}\left(\kappa^2 - 2\mathcal{D}_{\kappa\lambda}\hat{\mathcal{E}}(S)\mathcal{D}^{\kappa\lambda}\hat{\mathcal{E}}(S)\right), \quad \text{(E.29)}$$
$$G^0_{\mu\nu,\gamma\delta} = e^{-4\phi}\left(\kappa\delta_{\mu\nu} + 2\mathcal{D}_{\mu\nu}\hat{\mathcal{E}}(S)\right)\left(\frac{\pi}{6}\hat{\mathcal{G}}_{\gamma\delta}(\varphi) + 2\pi\delta_{\gamma\delta}(\hat{\mathcal{E}}(S) - 2\kappa\phi)\right),$$
$$G^0_{\alpha\beta,\gamma\delta} = e^{-4\phi}\left(\hat{\mathcal{G}}_{\alpha\beta,\gamma\delta}(\varphi) - \frac{\pi}{2}(\hat{\mathcal{E}}(S) - 2\kappa\phi)\delta_{\langle\alpha\beta},\hat{\mathcal{G}}_{\gamma\delta\rangle}(\varphi) - 3\pi\delta_{\langle\alpha\beta},\delta_{\gamma\delta\rangle}\left(\hat{\mathcal{E}}(S) - 2\kappa\phi\right)^2\right),$$

with

$$2\mathcal{D}_{(\gamma}{}^{\hat{\alpha}}\mathcal{D}_{\delta)\hat{\alpha}}\hat{\mathcal{G}}_{\alpha\beta} = -2\delta_{\gamma\delta}\hat{\mathcal{G}}_{\alpha\beta} - 2\delta_{\gamma)(\alpha}\hat{\mathcal{G}}_{\beta)(\delta} + \delta_{\alpha\beta}\hat{\mathcal{G}}_{\gamma\delta} + 12\hat{\mathcal{F}}_{\alpha\beta\gamma\delta} + 36\kappa\delta_{(\alpha\beta}\delta_{\gamma\delta)}\,,$$
$$2\mathcal{D}_{(\eta}{}^{\hat{\alpha}}\mathcal{D}_{\theta)\hat{\alpha}}\hat{\mathcal{G}}_{\alpha\beta,\gamma\delta} = -3\delta_{\eta\theta}\hat{\mathcal{G}}_{\alpha\beta,\gamma\delta} + 3\delta_{\langle\alpha\beta},\hat{\mathcal{G}}_{\gamma\delta\rangle,\eta\theta} - \frac{\kappa}{2}(\delta_{\eta\theta}\delta_{\langle\alpha\beta},\hat{\mathcal{G}}_{\gamma\delta\rangle} + 2\delta_{\eta\langle\alpha}\delta_{\beta,|\theta|}\hat{\mathcal{G}}_{\gamma\delta\rangle})$$
$$- 3\pi\hat{\mathcal{F}}_{\eta)\epsilon\langle\alpha\beta},\hat{\mathcal{F}}_{\gamma\delta\rangle(\theta}{}^\epsilon\,. \quad \text{(E.30)}$$

This is indeed consistent with the result (5.70) from the unfolding method, upon setting $\kappa = \frac{3}{\pi^2(N+1)} = \frac{k}{8\pi^2}$.

## E.3 Abelian Fourier coefficients

In this subsection we show that the generic Abelian Fourier coefficients of the tensor $G_{ab,cd}$ computed in section 5 satisfy the differential equation (E.11), including the quadratic source term.

For simplicity we shall only consider the component of the Fourier coefficient with all indices along the decompactified torus $G^{(p,q),\,2\text{Ab},(Q,P)}_{\mu\nu,\sigma\rho} = -\frac{1}{2}\varepsilon_{\mu(\sigma}\varepsilon_{\rho)\nu}G^{(p,q)}(Q,P)$. The latter is proportional to the scalar function

$$G^{(p,q)}(Q,P) = R^8 \sum_{\substack{A\in M_2(\mathbb{Z})/GL(2,\mathbb{Z}) \\ A^{-1}\binom{Q}{P}\in\Lambda^{\oplus 2}_{p-2,q-2}}} \int_{\mathcal{P}_2} \frac{d^3\Omega_2}{|\Omega_2|^{\frac{12-q}{2}}} |A|^2 C\big[A^{-1}\big(\begin{smallmatrix} Q^2 & Q\cdot P \\ Q\cdot P & P^2 \end{smallmatrix}\big)A^{-\intercal}; \Omega_2\big] L(A^{-\intercal}\Omega_2 A^{-1})\,, \qquad \text{(E.31)}$$

with

$$L(A^{-\intercal}\Omega_2 A^{-1}) = e^{-\pi R^2 \text{tr}\big[\nu A\Omega_2^{-1}A^\intercal \nu^\intercal\big] -2\pi\,\text{tr}\big[\Omega_2 A^{-1}\big(\begin{smallmatrix} Q_R^2 & Q_R\cdot P_R \\ Q_R\cdot P_R & P_R^2 \end{smallmatrix}\big)A^{-\intercal}\big]}\,. \qquad \text{(E.32)}$$

One can rewrite the differential operator in (E.11) as

$$\begin{aligned}
&\big(\mathcal{D}_\mu{}^{\hat{a}}\mathcal{D}_{\nu\hat{a}} + (q-5)\delta_{\mu\nu}\big)G^{(p,q)}(Q,P)e^{2\pi i(Qa^1+Pa^2)} \\
&= \Big(\big(\tfrac{1}{16}(-R\partial_R)^2 - \tfrac{q-1}{8}R\partial_R + q-5\big)\delta_{\mu\nu} - \tfrac{1}{2}\mathcal{D}_{\mu\nu}(-R\partial_R + q-2) + \mathcal{D}_{(\mu}{}^{\sigma}\mathcal{D}_{\nu)\sigma} \qquad \text{(E.33)} \\
&\qquad - 2\pi^2 R^2\big(\nu\big(\begin{smallmatrix} Q_R^2 & Q_R\cdot P_R \\ Q_R\cdot P_R & P_R^2 \end{smallmatrix}\big)\nu^\intercal\big)\Big)G^{(p,q)}(Q,P)e^{2\pi i(Qa^1+Pa^2)}\,.
\end{aligned}$$

Acting with this differential operator on $R^8 L(A^{-\intercal}\Omega_2 A^{-1})$ one obtains

$$\Big(\pi R^2(\nu A\Omega_2^{-1}A^\intercal \nu^\intercal)^2 + \tfrac{q-12}{2}\nu A\Omega_2^{-1}A^\intercal \nu^\intercal - 2\pi\nu\big(\begin{smallmatrix} Q_R^2 & Q_R\cdot P_R \\ Q_R\cdot P_R & P_R^2 \end{smallmatrix}\big)\nu^\intercal\Big)_{\mu\nu}\pi R^{10}|\Omega_2|^{\frac{q-12}{2}}L(A^{-\intercal}\Omega_2 A^{-1})\,, \qquad \text{(E.34)}$$

which allows to rewrite (E.34) as a total derivative in $\Omega_2$,

$$\begin{aligned}
&\big(\mathcal{D}_\mu{}^{\hat{a}}\mathcal{D}_{\nu\hat{a}} + (q-5)\delta_{\mu\nu}\big)R^8|\Omega_2|^{\frac{q-12}{2}}L(A^{-\intercal}\Omega_2 A^{-1})e^{2\pi i(Qa^1+Pa^2)} \\
&\qquad = (\pi R^2\nu A\frac{\partial}{\partial\Omega_2}A^\intercal \nu^\intercal)_{\mu\nu}R^8|\Omega_2|^{\frac{q-12}{2}}L(A^{-\intercal}\Omega_2 A^{-1})e^{2\pi i(Qa^1+Pa^2)}\,. \qquad \text{(E.35)}
\end{aligned}$$

By integration by parts, it follows that the Fourier coefficient would satisfy the homogeneous differential equation *if* the Fourier coefficient $C\big[A^{-1}\big(\begin{smallmatrix} Q^2 & Q\cdot P \\ Q\cdot P & P^2 \end{smallmatrix}\big)A^{-\intercal}; \Omega_2\big]$ did not depend on $\Omega_2$.

We shall now show that the dependence of the Fourier coefficients of $1/\Phi_{10}$ in $\Omega_2$, due to the poles at large $|\Omega_2|$ accounts for the appearance of the quadratic source term in the differential equation. Using (5.25) and (A.90), we obtain

$$\begin{aligned}
G^{(p,q)}(Q,P) &= R^8 \sum_{\substack{A\in M_2(\mathbb{Z})/GL(2,\mathbb{Z}) \\ A^{-1}\binom{Q}{P}\in\Lambda^{\oplus 2}_{p-2,q-2}}} \int_{\mathcal{P}_2} \frac{d^3\Omega_2}{|\Omega_2|^{\frac{12-q}{2}}} |A|^2 C^F\big[A^{-1}\big(\begin{smallmatrix} Q^2 & Q\cdot P \\ Q\cdot P & P^2 \end{smallmatrix}\big)A^{-\intercal}\big]L(A^{-\intercal}\Omega_2 A^{-1}) \\
&\quad + \frac{R^8}{2}\sum_{\substack{A\in M_2(\mathbb{Z})/\text{Dih}_4 \\ A^{-1}\binom{Q}{P}\in\Lambda^{\oplus 2}_{p-2,q-2}}} |A|^2 c\Big(-\frac{(A^{-1}\big(\begin{smallmatrix} Q^2 & Q\cdot P \\ Q\cdot P & P^2 \end{smallmatrix}\big)A^{-\intercal})_{11}}{2}\Big)c\Big(-\frac{(A^{-1}\big(\begin{smallmatrix} Q^2 & Q\cdot P \\ Q\cdot P & P^2 \end{smallmatrix}\big)A^{-\intercal})_{22}}{2}\Big) \\
&\quad \times \int_{\mathcal{P}_2}\frac{d^3\Omega_2}{|\Omega_2|^{\frac{12-q}{2}}}\Big(-\frac{1}{2\pi}\delta(\nu_2) - (A^{-1}\big(\begin{smallmatrix} Q^2 & Q\cdot P \\ Q\cdot P & P^2 \end{smallmatrix}\big)A^{-\intercal})_{12}\text{sign}(\nu_2) + \big|(A^{-1}\big(\begin{smallmatrix} Q^2 & Q\cdot P \\ Q\cdot P & P^2 \end{smallmatrix}\big)A^{-\intercal})_{12}\big|\Big)L(A^{-\intercal}\Omega_2 A^{-1}) \\
&\quad + \mathcal{O}(e^{-R^2})\,. \qquad \text{(E.36)}
\end{aligned}$$

The differential operator (E.35) annihilates the finite part of the Fourier coefficient, and gives

$$
\begin{aligned}
&\left(\mathcal{D}_\mu{}^{\hat{a}}\mathcal{D}_{\nu\hat{a}} + (q-5)\delta_{\mu\nu}\right)G^{(p,q)}(Q,P)e^{2\pi i(Qa^1 + Pa^2)} \\
&= -\frac{1}{2}\sum_{\substack{A\in M_2(\mathbb{Z})/\text{Dih}_4 \\ A^{-1}\binom{Q}{P}\in\Lambda^{\oplus 2}_{p-2,q-2}}} |A|^2 c\left(-\frac{(A^{-1}\left(\begin{smallmatrix} Q^2 & Q\cdot P \\ Q\cdot P & P^2\end{smallmatrix}\right)A^{-\mathsf{T}})_{11}}{2}\right)c\left(-\frac{(A^{-1}\left(\begin{smallmatrix} Q^2 & Q\cdot P \\ Q\cdot P & P^2\end{smallmatrix}\right)A^{-\mathsf{T}})_{22}}{2}\right) \\
&\quad\times\left(\frac{1}{2\pi}\left(\mathcal{D}_\mu{}^{\hat{a}}\mathcal{D}_{\nu\hat{a}} + (d-5)\delta_{\mu\nu}\right) - 2\pi R^2\left(\nu A\pi_{(1}A^{-1}\left(\begin{smallmatrix} Q^2 & Q\cdot P \\ Q\cdot P & P^2\end{smallmatrix}\right)A^{-\mathsf{T}}\pi_{2)}A^{\mathsf{T}}\nu^{\mathsf{T}}\right)_{\mu\nu}\right) \\
&\quad\times\int_0^\infty\frac{d\rho_2}{\rho_2^{\frac{12-q}{2}}}\int_0^\infty\frac{d\sigma_2}{\sigma_2^{\frac{12-q}{2}}}R^8 L(A^{-\mathsf{T}}\left(\begin{smallmatrix} \rho_2 & 0 \\ 0 & \sigma_2\end{smallmatrix}\right)A^{-1})e^{2\pi i(Qa^1 + Pa^2)} + \mathcal{O}(e^{-R^2}) \,, \quad (E.37)
\end{aligned}
$$

where the differential operator acting on the first term in (E.36) gives a total derivative, while the second term factorizes after integrating the Dirac delta function, and the third is integrated by part using $\frac{d}{d\Omega_2}\text{sign}(\text{tr}\left(\begin{smallmatrix} 0 & 1/2 \\ 1/2 & 0\end{smallmatrix}\right)\Omega_2) = \delta(\nu_2)\left(\begin{smallmatrix} 0 & 1 \\ 1 & 0\end{smallmatrix}\right)$ and

$$
\nu A\begin{pmatrix} 0 & 1 \\ 1 & 0\end{pmatrix}A^{\mathsf{T}}\nu^{\mathsf{T}}\left[A^{-1}\begin{pmatrix} Q^2 & Q\cdot P \\ Q\cdot P & P^2\end{pmatrix}A^{-\mathsf{T}}\right]_{12} = 2\nu A\pi_{(1}A^{-1}\begin{pmatrix} Q^2 & Q\cdot P \\ Q\cdot P & P^2\end{pmatrix}A^{-\mathsf{T}}\pi_{2)}A^{\mathsf{T}}\nu^{\mathsf{T}} \,. \quad (E.38)
$$

Further using (E.34) to express $\left(\mathcal{D}_\mu{}^{\hat{a}}\mathcal{D}_{\nu\hat{a}} + (d-5)\delta_{\mu\nu}\right)L(A^{-\mathsf{T}}\left(\begin{smallmatrix} \rho_2 & 0 \\ 0 & \sigma_2\end{smallmatrix}\right)A^{-1})e^{2\pi i(Qa^1 + Pa^2)}$, and inserting $\pi_1 + \pi_2 = \mathbb{1}$ on both sides of (E.35), we see that the terms which involve two powers of $\pi_1$ or two powers of $\pi_2$ cancel out since they are total derivatives with respect to $\rho_2$ or to $\sigma_2$, leaving only terms involving one factor of $\pi_1$ and one factor of $\pi_2$:

$$
\begin{aligned}
&\left(\mathcal{D}_\mu{}^{\hat{a}}\mathcal{D}_{\nu\hat{a}} + (q-5)\delta_{\mu\nu}\right)G^{(p,q)}(Q,P)e^{2\pi i(Qa^1 + Pa^2)} \\
&= -\frac{\pi}{2}\sum_{\substack{A\in M_2(\mathbb{Z})/\text{Dih}_4 \\ A^{-1}\binom{Q}{P}\in\Lambda^{\oplus 2}_{p-2,q-2}}} |A|^2 c\left(-\frac{(A^{-1}\left(\begin{smallmatrix} Q^2 & Q\cdot P \\ Q\cdot P & P^2\end{smallmatrix}\right)A^{-\mathsf{T}})_{11}}{2}\right)c\left(-\frac{(A^{-1}\left(\begin{smallmatrix} Q^2 & Q\cdot P \\ Q\cdot P & P^2\end{smallmatrix}\right)A^{-\mathsf{T}})_{22}}{2}\right)\int_0^\infty\frac{d\rho_2}{\rho_2^{\frac{12-q}{2}}}\int_0^\infty\frac{d\sigma_2}{\sigma_2^{\frac{12-q}{2}}} \\
&\quad\times\left(\nu A\pi_{(1}\left(\frac{R^2}{\sigma_2\rho_2}A^{\mathsf{T}}\nu^{\mathsf{T}}\nu A - 2A^{-1}\left(\begin{smallmatrix} Q_R^2 & Q_R\cdot P_R \\ Q_R\cdot P_R & P_R^2\end{smallmatrix}\right)A^{-\mathsf{T}} - 2A^{-1}\left(\begin{smallmatrix} Q^2 & Q\cdot P \\ Q\cdot P & P^2\end{smallmatrix}\right)A^{-\mathsf{T}}\right)\pi_{2)}A^{\mathsf{T}}\nu^{\mathsf{T}}\right)_{\mu\nu} \\
&\quad\times R^{10} L(A^{-\mathsf{T}}\left(\begin{smallmatrix} \rho_2 & 0 \\ 0 & \sigma_2\end{smallmatrix}\right)A^{-1})e^{2\pi i(Qa^1 + Pa^2)} + \mathcal{O}(e^{-R^2}) \\
&= -\frac{\pi}{2}\sum_{\substack{A\in M_2(\mathbb{Z})/\text{Dih}_4 \\ A^{-1}\binom{Q}{P}\in\Lambda^{\oplus 2}_{p-2,q-2}}} |A|^2 c\left(-\frac{(A^{-1}\left(\begin{smallmatrix} Q^2 & Q\cdot P \\ Q\cdot P & P^2\end{smallmatrix}\right)A^{-\mathsf{T}})_{11}}{2}\right)c\left(-\frac{(A^{-1}\left(\begin{smallmatrix} Q^2 & Q\cdot P \\ Q\cdot P & P^2\end{smallmatrix}\right)A^{-\mathsf{T}})_{22}}{2}\right) \\
&\quad\times\int_0^\infty\frac{d\rho_2}{\rho_2^{\frac{12-q}{2}}}\int_0^\infty\frac{d\sigma_2}{\sigma_2^{\frac{12-q}{2}}}\left(\nu A\pi_1\left(\frac{R^2}{\sigma_2\rho_2}A^{\mathsf{T}}\nu^{\mathsf{T}}\nu A - 2A^{-1}\left(\begin{smallmatrix} Q_L^2 & Q_L\cdot P_L \\ Q_L\cdot P_L & P_L^2\end{smallmatrix}\right)A^{-\mathsf{T}}\right)\pi_2 A^{\mathsf{T}}\nu^{\mathsf{T}}\right)_{(\mu\nu)} \\
&\quad\times R^{10} L(A^{-\mathsf{T}}\left(\begin{smallmatrix} \rho_2 & 0 \\ 0 & \sigma_2\end{smallmatrix}\right)A^{-1})e^{2\pi i(Qa^1 + Pa^2)} + \mathcal{O}(e^{-R^2}) \,, \quad (E.39)
\end{aligned}
$$

where in the last step we recognized $Q^2 + Q_R^2 = Q_L^2$. Defining for $i = 1$ or 2 the tensors

$$
L^i_{\mu\nu\sigma\rho}(\rho_2) = R^6(\nu A)_\mu{}^i(\nu A)_\nu{}^i(\nu A)_\sigma{}^i(\nu A)_\rho{}^i e^{-\frac{\pi}{\rho_2}R^2(\nu A)_\mu{}^i(\nu A)^{\mu i} - 2\pi\rho_2(A^{-1}\left(\begin{smallmatrix} Q_R^2 & Q_R\cdot P_R \\ Q_R\cdot P_R & P_R^2\end{smallmatrix}\right)A^{-\mathsf{T}})_{ii}} \quad (E.40)
$$

$$
L^i_{\mu\nu\sigma\alpha}(\rho_2) = iR^5(\nu A)_\mu{}^i(\nu A)_\nu{}^i(\nu A)_\sigma{}^i\left(A^{-1}\binom{Q_L}{P_L}\right)_{i\alpha} e^{-\frac{\pi}{\rho_2}R^2(\nu A)_\mu{}^i(\nu A)^{\mu i} - 2\pi\rho_2(A^{-1}\left(\begin{smallmatrix} Q_R^2 & Q_R\cdot P_R \\ Q_R\cdot P_R & P_R^2\end{smallmatrix}\right)A^{-\mathsf{T}})_{ii}}
$$

one obtains

$$
\begin{aligned}
&\left(\mathcal{D}_\mu{}^{\hat{a}}\mathcal{D}_{\nu\hat{a}}+(d-5)\delta_{\mu\nu}\right)G^{(p,q)}(Q,P)\,e^{2\pi i(Qa^1+Pa^2)}+\mathcal{O}(e^{-R^2})\\
&=\ -\frac{\pi}{2}\sum_{\substack{A\in M_2(\mathbb{Z})/\mathrm{Dih}_4\\ A^{-1}\binom{Q}{P}\in\Lambda^{\oplus 2}_{p-2,q-2}}}c\!\left(-\frac{\left(A^{-1}\big(\begin{smallmatrix}Q^2 & Q\cdot P\\ Q\cdot P & P^2\end{smallmatrix}\big)A^{-\intercal}\right)_{11}}{2}\right)c\!\left(-\frac{\left(A^{-1}\big(\begin{smallmatrix}Q^2 & Q\cdot P\\ Q\cdot P & P^2\end{smallmatrix}\big)A^{-\intercal}\right)_{22}}{2}\right)\varepsilon^{\sigma\kappa}\varepsilon^{\rho\lambda}e^{2\pi i(Qa^1+Pa^2)}\\
&\qquad\left(\int_0^\infty\frac{\mathrm{d}\rho_2}{\rho_2^{\frac{14-q}{2}}}L^1_{\sigma\rho\vartheta(\mu}(\rho_2)\int_0^\infty\frac{\mathrm{d}\sigma_2}{\sigma_2^{\frac{14-q}{2}}}L^2_{\nu)\kappa\lambda}{}^{\vartheta}(\sigma_2)+2\int_0^\infty\frac{\mathrm{d}\rho_2}{\rho_2^{\frac{12-q}{2}}}L^1_{\sigma\rho(\mu|a}(\rho_2)\int_0^\infty\frac{\mathrm{d}\sigma_2}{\sigma_2^{\frac{12-q}{2}}}L^2_{\nu)\kappa\lambda}{}^{a}(\sigma_2)\right)\\
&=\ -2\pi\varepsilon^{\sigma\kappa}\varepsilon^{\rho\lambda}\delta^{ab}e^{2\pi i(Qa^1+Pa^2)}\sum_{\substack{B\in M_2(\mathbb{Z})/\mathrm{Stab}\\ B\pi_1 B^{-1}\binom{Q}{P}\in\Lambda^{\oplus2}_{p-2,q-2}}}F^{(p,q),B\pi_1 B^{-1}\binom{Q}{P}}_{\sigma\rho(\mu|a}F^{(p,q),B\pi_2 B^{-1}\binom{Q}{P}}_{\nu)\kappa\lambda b}
\end{aligned}\tag{E.41}
$$

that indeed recovers (3.20),

$$
\begin{aligned}
\left(\mathcal{D}_\mu{}^{\hat{a}}\mathcal{D}_{\nu\hat{a}}+(d-5)\delta_{\mu\nu}\right)G^{(p,q)\Gamma}_{\sigma\rho,\kappa\lambda}&=\frac{\pi}{2}\varepsilon_{\kappa(\sigma}\varepsilon_{\rho)\lambda}\varepsilon^{\vartheta\upsilon}\varepsilon^{\tau\iota}\sum_{\Gamma_1\in\Lambda^*\oplus\Lambda}F^{(p,q)\Gamma_1}_{a\vartheta\tau(\mu}F^{(p,q)\Gamma-\Gamma_1 a}_{\nu)\upsilon\iota}\\
&=-\frac{3\pi}{2}\sum_{\Gamma_1\in\Lambda^*\oplus\Lambda}F^{(p,q)\Gamma_1}_{a\mu\langle\sigma\rho,}F^{(p,q)\Gamma-\Gamma_1 a}_{\kappa\lambda\rangle\nu}\,.
\end{aligned}\tag{E.42}
$$

Thus, we have shown that the abelian Fourier coefficients with generic 1/4-BPS charge are consistent with the differential constraint (3.20). This is a strong consistency check on the validity of the unfolding method in this sector.

## E.4 Differential equation in the degeneration $O(p,q)\to O(p-1,q-1)$

We now briefly discuss the consistency of the constant terms (4.20) computed in §4 with the differential equation (3.20). We follow [22, §B] for the parametrization of the Grassmannian and of the decomposition of the covariant derivative operators.

The operator $\mathcal{D}_{a\hat{b}}$ decomposes into $\mathcal{D}_{1\hat{1}}$, $\mathcal{D}_{1\hat{\beta}}$, $\mathcal{D}_{\alpha\hat{1}}$, $\mathcal{D}_{\alpha\hat{\beta}}$ acting on any vector $F_a=(F_1,F_\alpha)$,

$$
\begin{aligned}
\mathcal{D}_{1\hat{1}}F_a&=-\frac{1}{2}\frac{\partial}{\partial\phi}F_a\,,\\
\mathcal{D}_{\alpha\hat{1}}F_a&=\frac{1}{\sqrt{2}}e^{-\phi}v^{-1I}{}_\alpha\frac{\partial}{\partial a^I}F_a+\frac{1}{2}\left(F_\alpha,-\delta_{\alpha\beta}F_1\right)\,,\\
\mathcal{D}_{1\hat{\alpha}}F_a&=\frac{1}{\sqrt{2}}e^{-\phi}v^{-1I}{}_{\hat{\alpha}}\frac{\partial}{\partial a^I}F_b\,,
\end{aligned}\tag{E.43}
$$

and on any vector $F_{\hat{b}}=(F_{\hat{\beta}},F_{\hat{1}})$, as

$$
\begin{aligned}
\mathcal{D}_{1\hat{1}}F_{\hat{b}}&=-\frac{1}{2}\frac{\partial}{\partial\phi}F_{\hat{b}}\,,\\
\mathcal{D}_{\alpha\hat{1}}F_{\hat{b}}&=\frac{1}{\sqrt{2}}e^{-\phi}v^{-1I}{}_\alpha\frac{\partial}{\partial a^I}F_{\hat{b}}\,,\\
\mathcal{D}_{1\hat{\alpha}}F_{\hat{b}}&=\frac{1}{\sqrt{2}}e^{-\phi}v^{-1I}{}_{\hat{\alpha}}\frac{\partial}{\partial a^I}F_{\hat{b}}+\frac{1}{2}\left(-\delta_{\alpha\beta}F_{\hat{1}},F_{\hat{\alpha}}\right)\,,
\end{aligned}\tag{E.44}
$$

whereas $\mathcal{D}_{\alpha\hat{\beta}}$ reduce to the differential operators on the sub-Grassmannian that act on the projectors $p^I_{L\gamma}$, $p^I_{R\hat{\alpha}}$:

$$
\mathcal{D}_{\alpha\hat{\beta}}p^I_{L\gamma}=\tfrac{1}{2}\delta_{\alpha\gamma}p^I_{R\hat{\beta}}\,,\qquad\mathcal{D}_{\alpha\hat{\beta}}p^I_{L\hat{\alpha}}=\tfrac{1}{2}\delta_{\hat{\beta}\hat{\alpha}}p^I_{R\alpha}\,.\tag{E.45}
$$

In this decomposition, the tensor $G_{ab,cd}$ admits 3 independent components $G_{11,\gamma\delta}$, $G_{1\beta,\gamma\delta}$ and $G_{\alpha\beta,\gamma\delta}$, but only the first and last have a non-trivial constant term. Using the Fourier decomposition

$$G_{ab,cd} = \sum_{Q\in\Lambda^*} G^Q_{ab,cd}\, e^{2\pi i Q\cdot a}\,, \qquad F_{abcd} = \sum_{Q\in\Lambda^*} F^Q_{abcd}\, e^{2\pi i Q\cdot a}\,, \tag{E.46}$$

we obtain that the first component of (3.20) with $(e,f)=(1,1)$ reads

$$\big((\partial_\phi+4)(\partial_\phi+q-5)-8\pi^2 e^{-2\phi}Q_R^2\big)G^Q_{\alpha\beta,11} = -4\pi \sum_{Q_1\in\Lambda} \big(F^{Q_1}_{1k\alpha\beta}F^{Q-Q_1\,k}_{111} - F^{Q_1}_{11k(\alpha}F^{Q-Q_1\,k}_{\beta)11}\big),$$

$$\big((\partial_\phi+2)(\partial_\phi+q-3)-8\pi^2 e^{-2\phi}Q_R^2\big)G^Q_{\alpha\beta,\gamma\delta} = 6\delta_{\langle\alpha\beta,}G^Q_{\gamma\delta\rangle,11} - 6\pi \sum_{Q_1\in\Lambda} F^{Q_1}_{1k\langle\alpha\beta,}F^{Q-Q_1\,k}_{\gamma\delta\rangle 1}\,,$$
$$\tag{E.47}$$

where the sum over $k$ in the r.h.s. runs over all indices $\alpha$ and 1.

Introducing for brievity the vector $\vec{G}^Q$

$$\vec{G}^Q = (G^Q_{\alpha\beta,11}, G^Q_{\alpha\beta,\gamma\delta}),\tag{E.48}$$

the differential operator $\mathcal{D}^{\hat\alpha}_{(1}\mathcal{D}_{\eta)\hat\alpha}$ acts on $\vec{G}^Q$ as

$$2\mathcal{D}_1{}^{\hat c}\mathcal{D}_{\eta\hat c}\vec{G}_Q + 2\mathcal{D}_\eta{}^{\hat c}\mathcal{D}_{1\hat c}\vec{G}^Q = \mathcal{D}^{(Q)}_\eta\vec{G}^Q - (\partial_\phi+\tfrac{q-2}{2})\begin{pmatrix} 2G^Q_{1\eta,\gamma\delta} \\ -2\delta_{\eta(\alpha}G^Q_{1\beta),\gamma\delta}-2\delta_{\eta(\gamma}G^Q_{1\delta),\alpha\beta} \end{pmatrix}, \tag{E.49}$$

where we define for short

$$\mathcal{D}^{(Q)}_\eta \equiv -i\sqrt{2}e^{-\phi}\big(Q_{L\eta}(\partial_\phi+q-2)+2Q_{R\hat\alpha}\mathcal{D}^{\hat\alpha}_\eta\big).\tag{E.50}$$

The off-diagonal component of the differential equation (3.20) with $(e,f)=(1,\eta)$ take the form

$$\mathcal{D}^{(Q)}_\eta G^Q_{11,\gamma\delta} = 2(\partial_\phi+3)G^Q_{1\eta,\gamma\delta} - \pi \sum_{Q_1\in\Lambda^*}\big(F^{Q_1}_{111k}F^{Q-Q_1\,k}_{\eta\gamma\delta} + F^{Q_1}_{\eta11k}F^{Q-Q_1\,k}_{1\gamma\delta} - 2F^{Q_1}_{11k(\gamma}F^{Q-Q_1\,k}_{\delta)1\eta}\big),$$

$$\mathcal{D}^{(Q)}_\eta G^Q_{1\beta,\gamma\delta} = (\partial_\phi+2)G^Q_{\eta\beta,\gamma\delta} - (\partial_\phi+q-4)\big(\delta_{\eta\beta}G^Q_{\gamma\delta,11}-\delta_{\eta(\gamma}G^Q_{\delta)\beta,11}\big) - \delta_{\gamma\delta}G^Q_{\eta\beta,11}$$
$$+\delta_{\beta(\gamma}G^Q_{\delta)\eta,11} - 2\pi \sum_{Q_1\in\Lambda^*}\big(F^{Q_1}_{1k1\beta}F^{Q-Q_1\,k}_{\gamma\delta\eta} + F^{Q_1}_{1k\gamma\delta}F^{Q-Q_1\,k}_{1\beta\eta} - F^{Q_1}_{1k\beta(\gamma}F^{Q-Q_1\,k}_{\delta)1\eta} - F^{Q_1}_{1k1(\gamma}F^{Q-Q_1\,k}_{\delta)\beta\eta}\big),$$

$$\mathcal{D}^{(Q)}_\eta G^Q_{\alpha\beta,\gamma\delta} = -2(\partial_\phi+q-4)\big(\delta_{\eta(\alpha}G^Q_{\beta)1,\gamma\delta}+\delta_{\eta(\gamma}G^Q_{\delta)1,\alpha\beta}\big) + 3\delta_{\langle\alpha\beta}G^Q_{\gamma\delta\rangle,1\eta}$$
$$-6\pi \sum_{Q_1\in\Lambda^*} F^{Q_1}_{1k\langle\alpha\beta,}F^{Q-Q_1\,k}_{\gamma\delta\rangle\eta}.\tag{E.51}$$

The component of the differential operator with two indices along the subgrassmaniann $(e,f)=(\eta,\vartheta)$, acts as

$$4D_{(\eta}{}^{\hat c}D_{\theta)\hat c}\vec{G}^Q = \big(4\mathcal{D}_{(\eta}{}^{\hat\alpha}\mathcal{D}_{\theta)\hat\alpha} + \delta_{\eta\theta}\partial_\phi - 8\pi^2 e^{-2\phi}Q_{L\eta}Q_{L\theta}\big)\vec{G}^Q$$

$$+ 8\pi i\sqrt{2}e^{-\phi}Q_{L(\eta|}\begin{pmatrix} G^Q_{1|\theta),\gamma\delta} \\ -\delta_{|\theta)(\alpha}G^Q_{\beta)1,\gamma\delta}-\delta_{|\theta)(\gamma}G^Q_{\delta)1,\alpha\beta} \end{pmatrix}\tag{E.52}$$

$$+\begin{pmatrix} 2G^Q_{\eta\theta,\gamma\delta}-2\delta_{\eta\theta}G^Q_{11,\gamma\delta}+2\delta_{|\eta)(\gamma}G^Q_{\delta)(\theta|,11} \\ 6\delta_{\eta\langle(\alpha}\delta_{\beta),|\theta|}G^Q_{\gamma\delta\rangle,11}-4\delta_{(\eta|\langle(\alpha}G^Q_{\beta),|\theta),|\gamma\delta\rangle} \end{pmatrix},$$

and thus we obtain the differential equation on the sub-Grassmaniann $G_{p-2,q-2}$

$$\left(2\mathcal{D}_{(\eta}{}^{\hat{\alpha}}\mathcal{D}_{\theta)\hat{\alpha}} + \tfrac{1}{2}\delta_{\eta\theta}\partial_{\phi} - 4\pi^2 e^{-2\phi}Q_{L\eta}Q_{L\theta}\right)\vec{\mathcal{G}}^Q$$

$$+ 4\pi\mathrm{i}\sqrt{2}e^{-\phi}Q_{L(\eta}\begin{pmatrix} G^Q_{\theta)1,\gamma\delta} \\ -\delta_{\theta)(\alpha}G^Q_{\beta)1,\gamma\delta} - \delta_{\theta)(\gamma}G^Q_{\delta)1,\alpha\beta} \end{pmatrix}$$

$$= \begin{pmatrix} (4-q)\delta_{\eta\theta}G^Q_{11,\gamma\delta} + (5-q)\delta_{|\eta)(\gamma}G^Q_{\delta)(\theta|,11} + \delta_{\gamma\delta}G^Q_{\eta\theta,11} \\ (3-q)\delta_{\eta\theta}G^Q_{\alpha\beta,\gamma\delta} + 2(6-q)\delta_{|\eta)((\alpha}G^Q_{\beta),(\theta|,|\gamma\delta)} + 3\delta_{\langle\alpha\beta,}G^Q_{\gamma\delta\rangle,\eta\theta} - 3\delta_{\eta\langle(\alpha}\delta_{\beta),|\theta|}G^Q_{\gamma\delta\rangle,11} \end{pmatrix}$$

$$- 2\pi \sum_{Q_1\in\Lambda^*}\begin{pmatrix} F^{Q_1}_{11d(\eta}F^{Q-Q_1 d}_{\theta)\gamma\delta} - F^{Q_1}_{1\gamma d(\eta}F^{Q-Q_1 d}_{\theta)\delta 1} \\ \tfrac{3}{2}F^{Q_1}_{\eta d\langle\alpha\beta,}F^{Q-Q_1 d}_{\gamma\delta\rangle\theta} \end{pmatrix}. \tag{E.53}$$

The constant terms sourcing the perturbative part of $G^0_{ab,cd}$ are given by [22, (4.16)]

$$\begin{aligned} F^0_{1111} &= a\, e^{-(q-6)\phi}\xi(q-6)\,, \\ F^0_{11\gamma\delta} &= b\, e^{-(q-6)\phi}\xi(q-6)\delta_{\delta\gamma}\,, \\ F^0_{\alpha\beta\gamma\delta} &= e^{-\phi}\mathcal{F}_{\alpha\beta\gamma\delta} + c\, e^{-(q-6)\phi}\xi(q-6)\delta_{(\alpha\beta}\delta_{\gamma\delta)}\,, \end{aligned} \tag{E.54}$$

with $a$, $b$, $c$ constants which were computed in [22] ($a = \frac{3(7-q)(9-q)}{\pi^2}$, $b = \frac{3(7-q)}{\pi^2}$, $c = \frac{9}{\pi^2}$). As mentioned in the previous section, the unfolding method fails to capture exponentially suppressed corrections to the constant term, which are sourced by the instanton anti-instanton quadratic terms $\sum_{Q_1\neq 0}F^{Q_1}F^{-Q_1}$ on the right-hand side of the differential equations (E.47), (E.51), (E.53). These terms can in principle be computed by solving the differential equation. Here we focus on the perturbative part sourced by the constant terms in (E.54).

We find the particular solution to equations (E.53)

$$G^0_{11,\gamma\delta} = -\frac{\pi c}{18}e^{-(q-5)\phi}\xi(q-6)(7-q)\mathcal{G}_{\gamma\delta}(\varphi) - \frac{2\pi c^2}{9}(7-q)e^{-2(q-6)\phi}\xi(q-6)^2\delta_{\alpha\beta}\,,$$

$$G^0_{\alpha\beta,\gamma\delta} = e^{-2\phi}\mathcal{G}_{\alpha\beta,\gamma\delta}(\varphi) - \frac{\pi c}{6}e^{-(q-5)\phi}\xi(q-6)\,\delta_{\langle\alpha\beta,}\mathcal{G}_{\gamma\delta\rangle}(\varphi) - \frac{\pi c^2}{3}e^{-2(q-6)\phi}\xi(q-6)^2\delta_{\langle\alpha\beta}\delta_{\gamma\delta\rangle}\,, \tag{E.55}$$

and $b = \frac{7-q}{3}c$ in (E.54), which matches the result obtained in [22, (4.16)]. $\mathcal{G}_{\alpha\beta,\gamma\delta}(\varphi)$ is a solution to the equation (3.20) on the sub-Grassmaniann $G_{p-1,q-1}$ with source term quadratic in $\mathcal{F}_{\alpha\beta\gamma\delta}(\varphi)$, and $\mathcal{G}_{\alpha\beta}(\varphi)$ satisfies the equation (B.18) along $G_{p-1,q-1}$

$$2\mathcal{D}_{(\eta}{}^{\hat{\alpha}}\mathcal{D}_{\theta)\hat{\alpha}}\mathcal{G}_{\alpha\beta} = \frac{3-q}{2}\delta_{\eta\theta}\mathcal{G}_{\alpha\beta} + (5-q)\delta_{|\eta)(\alpha}\mathcal{G}_{\beta)(\theta|} + \delta_{\alpha\beta}\mathcal{G}_{\eta\theta} + 12\mathcal{F}_{\alpha\beta\eta\theta}\,. \tag{E.56}$$

One can check that $G^0_{ab,cd}$ is also solution to (E.47), setting $a = \frac{(7-q)(9-q)c}{3}$ which matches the results [22, (4.16)].

The most general solution to (E.53) is obtained by adding a solution of the homogeneous equation without source term

$$\begin{aligned} \tilde{G}^0_{11,\gamma\delta} &= -\frac{\pi c}{18}(7-q)e^{-(q-5)\phi}\xi(q-6)\tilde{\mathcal{G}}_{\gamma\delta}(\varphi)\,, \\ \tilde{G}^0_{\alpha\beta,\gamma\delta} &= e^{-2\phi}\tilde{\mathcal{G}}_{\alpha\beta,\gamma\delta}(\varphi) - \frac{\pi c}{6}e^{-(q-5)\phi}\xi(q-6)\,\delta_{\langle\alpha\beta,}\tilde{\mathcal{G}}_{\gamma\delta\rangle}(\varphi)\,, \end{aligned} \tag{E.57}$$

with $\tilde{\mathcal{G}}_{\alpha\beta,\gamma\delta}(\varphi)$ a solution to the homogeneous equation (3.17) on the sub-Grassmaniann $G_{p-1,q-1}$, and $\tilde{\mathcal{G}}_{\alpha\beta}(\varphi)$ solution to the homogeneous equation (3.34) on $G_{p-1,q-1}$. The results

(4.59) and (4.60) for the constant term $G^0_{ab,cd}$ obtained by the unfolding method in this decomposition lie in this class, upon setting for generic $q$

$$
\begin{aligned}
\mathcal{G}_{\alpha\beta} &= \frac{\upsilon N^{q-7}+1}{N+1} \tfrac{1}{2}(G^{(p-1,q-1)}_{\alpha\beta}(\varphi) + {}^\varsigma G^{(p-1,q-1)}_{\alpha\beta}(\varphi)), \\
\tilde{\mathcal{G}}_{\alpha\beta} &= \frac{\upsilon N^{q-7}-1}{N-1} \tfrac{1}{2}(G^{(p-1,q-1)}_{\alpha\beta}(\varphi) - {}^\varsigma G^{(p-1,q-1)}_{\alpha\beta}(\varphi)),
\end{aligned}
\tag{E.58}
$$

with $c = \frac{9}{\pi^2}$ as in [22].

## F  Beyond the saddle point approximation

In the analysis of the large radius limit of the genus-two modular integral in §5.1, we neglected the dependence of the Fourier coefficients $C_{k-2}(n,m,L;\Omega_2)$ of the meromorphic Siegel modular form $1/\Phi_{k-2}$ on $\Omega_2$, and evaluated the integral over $\Omega_2$ arising in the Abelian rank-two orbit in terms of a matrix variate Bessel function. Since the integral over $\Omega_2$ is dominated by a saddle point at large $R$, and since $C_{k-2}(n,m,L;\Omega_2)$ is constant in the vicinity of the saddle point (at least at generic point in the moduli space $G_4/K_4$), this approximation correctly captures the leading behavior of order $e^{-2\pi R\mathcal{M}(Q,P)}$ at large $R$, as well as the infinite series of perturbative corrections around the saddle point. As a result of the poles in $1/\Phi_{k-2}$ however, the Fourier coefficient $C_{k-2}(n,m,L;\Omega_2)$ is only locally constant, and this approximation misses contributions from the region where this Fourier coefficient differs from its saddle point value. Here we shall estimate these effects and find that

1. poles occuring at large $|\Omega_2|$ give rise to contributions of order $e^{-2\pi R(\mathcal{M}(Q_1,P_1)+\mathcal{M}(Q_2,P_2))}$ for all possible splittings $(Q,P) \to (Q_1,P_1) + (Q_2,P_2)$ of the total charge into a pair of 1/2-BPS charges; these contributions are subleading away from the walls of marginal stability, but crucial for the smoothness of the physical couplings across the wall;

2. deep poles occuring at $|\Omega_2| \leq \frac{1}{(2n_2)^2}$ give rises to subleading contributions exponentially suppressed in $e^{-4\pi|n_2|R^2}$ which can be interpreted as $|n_2|$ pairs of Taub-NUT instantons and anti-instantons.

In either case, the gist of the argument is as follows: one decomposes the integral

$$
\int_{\mathcal{P}_2} \frac{d^3\Omega_2}{|\Omega_2|^{\frac{3}{2}-s}} C[\Omega_2]e^{-2\pi S[\Omega_2]} = \sum_k C_k \int_{W_k} \frac{d^3\Omega_2}{|\Omega_2|^{\frac{3}{2}-s}} e^{-2\pi S[\Omega_2]} \,,
\tag{F.1}
$$

with a locally constant insertion $C[\Omega_2]$ into a sum over chambers $W_k$ where $C[\Omega_2] = C_k$ is constant. Applying the saddle point approximation, one can bound the integral over $W_k$ at large $R$ as

$$
-\frac{1}{2\pi} \log\!\left( \int_{W_k} \frac{d^3\Omega_2}{|\Omega_2|^{\frac{3}{2}-s}} e^{-2\pi S[\Omega_2]} \right) = S[\Omega_2^*(W_k)] + o(R) \,,
\tag{F.2}
$$

where $\Omega_2^*(W_k)$ is the minimum of $S[\Omega]$ on $W_k$.

### F.1  Poles at large $|\Omega_2|$

Recall that the saddle point lies at

$$
\Omega_2^* = \frac{R}{\mathcal{M}(Q,P)} A^{\intercal} \left[ \frac{1}{S_2}\begin{pmatrix} 1 & S_1 \\ S_1 & |S|^2 \end{pmatrix} + \frac{1}{|Q_R \wedge P_R|}\begin{pmatrix} P_R^2 & -Q_R \cdot P_R \\ -Q_R \cdot P_R & Q_R^2 \end{pmatrix} \right] A \,,
\tag{F.3}
$$

where $A$ is a non-generate integer matrix, which we decompose as $A = \begin{pmatrix} 1 & \frac{j}{k} \\ 0 & 1 \end{pmatrix}\begin{pmatrix} d_1 & 0 \\ 0 & d_2 \end{pmatrix}\gamma$ for $\gamma \in SL(2,\mathbb{Z})$. We consider the component of the diagonal divisor $\mathcal{D}$ where the matrix $\begin{pmatrix} 1 & 0 \\ \frac{j}{k} & 1 \end{pmatrix}A^{-\intercal}\Omega_2 A^{-1}\begin{pmatrix} 1 & \frac{j}{k} \\ 0 & 1 \end{pmatrix}$ becomes diagonal. On this divisor, we parametrize $\Omega_2$ as

$$\Omega_2 = RA^{\intercal}\begin{pmatrix} 1 & 0 \\ -\frac{j}{k} & 1 \end{pmatrix}\begin{pmatrix} \rho_2 & 0 \\ 0 & \sigma_2 \end{pmatrix}\begin{pmatrix} 1 & -\frac{j}{k} \\ 0 & 1 \end{pmatrix}A\,. \tag{F.4}$$

It is straightforward to compute the minimum of the action

$$S[\Omega_2] = \frac{R^2}{2}\,\mathrm{tr}\Big[\Omega_2^{-1}A^{\intercal}\frac{1}{S_2}\begin{pmatrix} 1 & S_1 \\ S_1 & |S|^2 \end{pmatrix}A\Big] + \mathrm{tr}\Big[\Omega_2 A^{-1}\begin{pmatrix} Q_R^2 & Q_R\cdot P_R \\ Q_R\cdot P_R & P_R^2 \end{pmatrix}A^{-\intercal}\Big] \tag{F.5}$$

on the surface parametrized by $\sigma_2$ and $\rho_2$, because the matrices in the traces are then diagonal. The minimum is reached at

$$\Omega_2' = RA^{\intercal}\begin{pmatrix} 1 & 0 \\ -\frac{j}{k} & 1 \end{pmatrix}\begin{pmatrix} \frac{1}{\sqrt{2S_2(Q_R-\frac{j}{k}P_R)^2}} & 0 \\ 0 & \frac{|S+\frac{j}{k}|}{\sqrt{2S_2 P_R^2}} \end{pmatrix}\begin{pmatrix} 1 & -\frac{j}{k} \\ 0 & 1 \end{pmatrix}A\,, \tag{F.6}$$

with

$$S[\Omega_2'] = R\Big(\sqrt{\tfrac{2}{S_2}(Q_R-\tfrac{j}{k}P_R)^2} + \sqrt{\tfrac{2|S+\frac{j}{k}|^2}{S_2}P_R^2}\Big) = R\big(\mathcal{M}(Q-\tfrac{j}{k}P,0) + \mathcal{M}(\tfrac{j}{k}P,P)\big)\,, \tag{F.7}$$

which we recognize as the sums of the actions associated to 1/2-BPS states with charge $(Q_1,P_1) = (Q-\frac{j}{k}P,0)$ and $(Q_2,P_2) = (\frac{j}{k}P,P)$, as announced. Taking $j=0$ for simplicity and parametrizing the distance away from the divisor $\nu = 0$ by $\epsilon$ such that

$$\frac{S_1}{S_2} - \frac{Q_R\cdot P_R}{|Q_R\wedge P_R|} = \epsilon \qquad \Rightarrow \qquad Q_R\cdot P_R = \sqrt{\frac{Q_R^2 P_R^2}{S_2^2 + (S_1-\epsilon S_2)^2}}(S_1-\epsilon S_2)\,, \tag{F.8}$$

the perturbation of the action at small $\nu_2$ gives

$$\begin{aligned} S\Big[RA^{\intercal}\begin{pmatrix} \frac{1}{\sqrt{2S_2 Q_R^2}} & \nu_2 \\ \nu_2 & \frac{|S|}{\sqrt{2S_2 P_R^2}} \end{pmatrix}A\Big] &= S[\Omega_2'] + 2R\nu_2\sqrt{Q_R^2 P_R^2}\Big(\frac{S_1-\epsilon S_2}{\sqrt{S_2^2+(S_1-\epsilon S_2)^2}} - \frac{S_1}{|S|}\Big) + \mathcal{O}(\nu_2^2) \\ &= S[\Omega_2'] - 2R\nu_2\sqrt{Q_R^2 P_R^2}\frac{S_2^3}{|S|^3}\epsilon + \mathcal{O}(\epsilon^2) + \mathcal{O}(\nu_2^2)\,. \end{aligned} \tag{F.9}$$

For $\epsilon$ small enough, *i.e.* $\Omega_2^*$ close enough to the wall $\nu_2 = 0$, one sees indeed that the action increases monotonically away from the wall, and therefore, the minimum of the action in the neighboring chamber must indeed be reached along the wall. All the other cases are then determined from this one by $SL(2,\mathbb{Z})$.

## F.2 Deep poles

When the determinant $|\Omega_2|$ becomes sufficiently small, the contour $\mathcal{C} = [0,1]^3 + i\Omega_2$ starts intersecting additional poles of the form

$$m^2 - m^1\rho + n_1\sigma + n_2(\rho\sigma - \nu^2) + j\nu = 0, \tag{F.10}$$

with $n_2 \neq 0$, $4(m^1 n_1 + m^2 n_2) + j^2 = 1$. This intersection occurs for generic values of $\Omega_2$, which make the Fourier coefficient $C(m,n,p;\Omega_2)$ itself ill-defined. In this section it is convenient to parametrize $\Omega_2$ as

$$\Omega_2 = \frac{1}{V\tau_2}\begin{pmatrix} 1 & \tau_1 \\ \tau_1 & |\tau|^2 \end{pmatrix}\,. \tag{F.11}$$

Eq. (F.10) can be solved for $\tau_1, v_1$ as a function of $\tau_2, V, \rho_1, \sigma_1$,

$$
v_1 = \frac{1}{2n_2} \left( j \pm \sqrt{1 - 4(n_1 + n_2\rho_1)(m_1 - n_2\sigma_1) - \frac{4n_2^2}{V^2}} \right),
$$

$$
\tau_1 = \frac{1}{2(n_1 + n_2\rho_1)} \left[ \sqrt{1 - 4(n_1 + n_2\rho_1)(m_1 - n_2\sigma_1) - \frac{4n_2^2}{V^2}} - \sqrt{1 - 4(n_1 + n_2\rho_1)^2\tau_2^2 - \frac{4n_2^2}{V^2}} \right].
$$
(F.12)

The solution is real only if $V^2 - 4V^2(n_1 + n_2\rho_1)^2\tau_2^2 - 4n_2^2 \geq 0$, which requires $V^2 \geq 4n_2^2$, *i.e.* that $|\Omega_2| < 1/(2n_2)^2$.

In order to bound the contribution from this region, we shall look for the minimum of the action (F.5) on the domain $\mathcal{P}_2$ with $|\Omega_2| < \frac{1}{(2n_2)^2}$. For simplicity we consider the case $A = 1$, but the argument is general. Extremizing over $\tau$ in the parametrization (F.11) one obtains the solution

$$
\tau^* = S_1 + S_2 \frac{-P_R \cdot (Q_R + S_1 P_R) + i\sqrt{|Q_R \wedge P_R|^2 + \frac{R^2 V^2}{2} \frac{|Q_R + S P_R|^2}{S_2} + \frac{R^4 V^4}{4}}}{P_R^2 + \frac{R^2 V^2}{2}},
$$
(F.13)

at which point the action becomes

$$
S[\tau^*, V] = \sqrt{R^4 V^2 + 2R^2 \frac{|Q_R + S P_R|^2}{S_2} + \frac{4}{V^2} |Q_R \wedge P_R|^2}.
$$
(F.14)

At large $R$ the action grows monotonically in $V$, so the minimum of the action on the domain $V \geq 2|n_2|$ is reached on the boundary at $V = 2|n_2|$, where it evaluates to

$$
S[\tau^*, 2|n_2|] = \sqrt{(2n_2 R^2)^2 + 2R^2 \frac{|Q_R + S P_R|^2}{S_2} + \frac{1}{n_2^2} |Q_R \wedge P_R|^2}.
$$
(F.15)

The correction in this domain are therefore exponentially suppressed as $e^{-4\pi R^2|n_2|}$, which is the expected magnitude of a contribution for $|n_2|$ pairs of Taub-NUT instanton anti-instantons.

## G  Non-Abelian Fourier coefficients

In this section we show that the non-Abelian Fourier coefficients in the degeneration $(p, q) \to (p - 2, q - 2)$ can be deduced from the (Abelian) Fourier coefficients in the degeneration $(p, q) \to (p - 1, q - 1)$. First, recall that the Fourier expansion of an automorphic form $F$ on $G_{p,q}$ with respect to the maximal parabolic subgroup with Levi $GL(1) \times O(p - 2, q - 2)$, corresponding to the grading (5.3), which we copy for convenience,

$$
\mathfrak{so}_{p,q} \simeq \ldots \oplus (\mathfrak{gl}_1 \oplus \mathfrak{sl}_2 \oplus \mathfrak{so}_{p-2,q-2})^{(0)} \oplus (\mathbf{2} \otimes (\mathbf{p+q-4}))^{(1)} \oplus \mathbf{1}^{(2)},
$$
(G.1)

consists of three parts:

1.  the constant term $F_0(R, t)$, defined as the average of $F$ with respect to $(a^{iI}, \psi)$ parametrizing the grade (1) and (2) components in (G.1);

2.  the Abelian Fourier coefficients $F_{Q,P}(R, t)$, defined as the average of the product of $F$ by a character $e^{-2\pi i(Qa_1 + Pa_2)}$ with $(Q, P)$ in the lattice $\Lambda_{p-2,q-2}^{\oplus 2}$;

3. the non-Abelian Fourier coefficients $F_{M_1}(R, t, a)$ for $M_1 \in \mathbb{Z}\backslash\{0\}$, defined as the average of $F$ times $e^{-2\pi i M_1 \psi}$ over $\psi \in [0, 1]$.

The non-Abelian Fourier coefficient can be further decomposed by diagonalizing a half dimensional Lagrangian subspace in the grade (1) space, e.g. dual to the lattice $\Lambda_{p-2,q-2}$ of magnetic charges. This leads to the 'wave function representation'

$$F_{M_1}(R, t, a^1, a^2) = \sum_{\mu \in \frac{\Lambda_{p-2,q-2}}{M_1 \Lambda_{p-2,q-2}}} \sum_{P \in M_1 \Lambda_{p-2,q-2}+\mu} F_{M_1,\mu}(R, t; P - M_1 a_1) e^{2\pi i(P \cdot a_2 - \frac{1}{2} M_1 a_1 \cdot a_2)}. \quad (G.2)$$

However, an alternative representation of the same non-Abelian Fourier coefficient can be obtained by diagonalizing not only translations in $\psi$ and $a_2$ but also in $S_1$, corresponding to the positive root in the $\mathfrak{sl}(2)$ factor appearing in the grade 0 component of (G.1). This amounts to performing the (Abelian) Fourier decomposition with respect to a different maximal parabolic subgroup associated to the decomposition (4.4),

$$\mathfrak{so}_{p,q} \simeq (\mathbf{p+q-2})^{(-2)} \oplus (\mathfrak{gl}_1 \oplus \mathfrak{so}_{p-1,q-1})^{(0)} \oplus (\mathbf{p+q-2})^{(2)}. \quad (G.3)$$

The only task is to relate the coordinates $(R, S, \varphi, a^1, a^2)$ appropriate to (G.1) to the coordinates $(R', \varphi', a')$ appropriate to (G.3). To this aim, let us parametrize the $(SO(p) \times SO(q))\backslash SO(p,q)$ Grassmannian in the parabolic gauge as

$$g(R, S_2, S_1, \varphi, a_1, a_2, \psi) = L(R, S_2, \varphi) U_2(S_1) U_{\text{e.m.}}(a_1, a_2) U_1(\psi), \quad (G.4)$$

with $L(1, 1, \varphi) \subset SO(p-2, q-2)$, $L(1, S_2, 0) U_2(S_1) \subset SL(2, \mathbb{R})$ and $U_{\text{e.m.}}(a_1, a_2) U_1(\psi)$ in the unipotent radical. One straightforwardly computes that

$$\begin{aligned}
&\left[ L(R, S_2, \varphi) U_2(S_1) \right] U_{\text{e.m.}}(a_1, a_2) U_1(\psi) && (G.5) \\
=\ & L(R, S_2, \varphi) U_2(S_1) U_{\text{e.m.}}(a_1, 0) U_{\text{e.m.}}(0, a_2) U_1(\psi - \tfrac{1}{2} a_1 \cdot a_2) \\
=\ & \left[ L(R, S_2, \varphi) U_{\text{e.m.}}(a_1, 0) \right] U_2(S_1) U_{\text{e.m.}}(0, a_2 - S_1 a_1) U_1(\psi - \tfrac{1}{2} a_1 \cdot a_2 + \tfrac{1}{2} S_1 a_1^2),
\end{aligned}$$

where $L(R, S_2, \varphi) U_{\text{e.m.}}(a_1, 0) \in \mathbb{R}^+ \times SO(p-1, q-1)$ and $U_2(S_1) U_{\text{e.m.}}(0, a') U_1(\psi')$ belongs to the corresponding abelian unipotent radical. Using this parametrization, the non-abelian Fourier coefficients can simply be obtained from the Fourier coefficients (4.44) by substituting

$$\begin{aligned}
R' &= R\sqrt{S_2}, \\
Q'_{R\hat{\alpha}} &= \begin{cases} \frac{1}{\sqrt{2S_2}} \left( RM_1 + \frac{S_2}{R}(M_2 - a_1 \cdot P + \frac{1}{2} a_1^2 M_1) \right) & \text{if } \hat{\alpha} = q-1 \\ P_{R\hat{\alpha}} - a_{1R\hat{\alpha}} M_1 & \text{if } \hat{\alpha} < q-1 \end{cases}, \\
Q'_{L\alpha} &= \begin{cases} \frac{1}{\sqrt{2S_2}} \left( RM_1 - \frac{S_2}{R}(M_2 - a_1 \cdot P + \frac{1}{2} a_1^2 M_1) \right) & \text{if } \alpha = p-1 \\ P_{L\alpha} - a_{1L\alpha} M_1 & \text{if } \alpha < p-1 \end{cases}, \\
Q \cdot a' &\rightarrow M_1(\psi - \tfrac{1}{2} a_1 \cdot a_2 + \tfrac{1}{2} S_1 a_1^2) + P \cdot (a_2 - S_1 a_1) + M_2 S_1, \quad (G.6)
\end{aligned}$$

where $Q = (P, M_1, M_2) \in \Lambda_{p-1,q-1}$ split into $P \in \Lambda_{p-2,q-2}$ and $(M_1, M_2) \in \mathrm{I\!I}_{1,1}$, such that $Q^2 = P^2 - 2M_1 M_2$. The index $\mu = 1$ is combined with the index $\alpha$ ranging from 1 to $p-2$ of $SO(p-2)$ to give the index $\alpha$ ranging from 1 to $p-1$ of $SO(p-1)$, whereas the index $\mu = 2$ corresponds to the index 1 in the decomposition (4.44). The non-Abelian Fourier coefficient $F_{M_1}(R, t, a^1, a^2)$ of $G_{abcd}^{(p,q)}$ is then

$$G_{ab,cd}^{(p,q),\,2\text{nab},M_1} = \sum_{P \in \Lambda_{p-2,q-2}} \left( \sum_{M_2 \in \mathbb{Z}} G_{ab,cd}^{(p,q),(P,M_1,M_2)} e^{2\pi i(M_2 - a_1 \cdot P + \frac{1}{2} a_1^2 M_1) S_1} \right) e^{2\pi i(P \cdot a_2 - \frac{1}{2} M_1 a_1 \cdot a_2)}, \quad (G.7)$$

with the classical action

$$S_{\mathrm{cl}}(M_1, M_2, P) = \sqrt{\left(R^2 M_1 + S_2(M_2 - a_1 \cdot P + \tfrac{1}{2}a_1^2 M_1)\right)^2 + 2R^2 S_2(P_R - a_{1R}M_1)^2}, \qquad \text{(G.8)}$$

and

$$G_{\alpha\beta,\gamma\delta}^{(p,q),(P,M_1,M_2)} = 6\,\bar{G}_{\langle\alpha\beta,}^{(p-1,q-1)}(P,M_1,M_2;\tfrac{R^2}{S_2},\varphi,a_1) \sum_{l=0}^{1} \frac{\tilde{P}_{\gamma\delta\rangle}^{(l)}(P_L - a_{1L}M_1)}{(R^2 S_2)^{l-\frac{q-2}{2}}} \frac{K_{\frac{q-5}{2}-\ell}(2\pi S_{\mathrm{cl}})}{S_{\mathrm{cl}}^{\frac{q-3}{2}-l}},$$

$$G_{\alpha\beta,\gamma2}^{(p,q),(P,M_1,M_2)} = 3(R^2 S_2)^{\frac{q-3}{2}}\,\bar{G}_{\langle\alpha\beta,}^{(p-1,q-1)}(P,M_1,M_2;\tfrac{R^2}{S_2},\varphi,a_1) \frac{P_{L\gamma} - a_{1L\gamma\rangle}M_1}{i\sqrt{2}} \frac{K_{\frac{q-7}{2}}(2\pi S_{\mathrm{cl}})}{S_{\mathrm{cl}}^{\frac{q-5}{2}}},$$

$$G_{\alpha\beta,22}^{(p,q),(P,M_1,M_2)} = -(R^2 S_2)^{\frac{q-4}{2}}\,\bar{G}_{\alpha\beta}^{(p-1,q-1)}(P,M_1,M_2;\tfrac{R^2}{S_2},\varphi,a_1) \frac{K_{\frac{q-9}{2}}(2\pi S_{\mathrm{cl}})}{S_{\mathrm{cl}}^{\frac{q-7}{2}}}, \qquad \text{(G.9)}$$

whereas the components with $\mu = 1$ are obtained by replacing $Q'_{L\alpha} = P_L - a_{1L}M_1$ for $\alpha = 1$, $p-2$ by $Q'_{Lp-1}$ in (G.6). The tensor $\bar{G}_{\alpha\beta}^{(p-1,q-1)}(P,M_1,M_2)$ is defined on $SO(p-1,q-1)$ from (4.45) in the parabolic gauge in which $g(\tfrac{R^2}{S_2},t,a_1) = L((\tfrac{R^2}{S_2})^{\frac{1}{4}},(\tfrac{S_2}{R^2})^{\frac{1}{2}},\varphi)U_{\mathrm{e.m.}}(a_1,0)$, with $Q = (P,M_1,M_2)$, and with the index $\alpha = p-1$ interpreted as $\mu = 1$, according to (G.6). Note that the tensor $\bar{G}_{\alpha\beta}^{(p-1,q-1)}(P,M_1,M_2)$ is not invariant under the shift of $a_1$ by a vector $\epsilon \in \Lambda_{p-2,q-2}$, but satisfies

$$\bar{G}_{\alpha\beta}^{(p-1,q-1)}(P,M_1,M_2;\tfrac{R^2}{S_2},t,a_1+\epsilon) = \bar{G}_{\alpha\beta}^{(p-1,q-1)}(P-\epsilon M_1,M_1,M_2-\epsilon\cdot P + \tfrac{1}{2}\epsilon^2 M_1;\tfrac{R^2}{S_2},\varphi,a_1), \quad \text{(G.10)}$$

which ensures that the decomposition (G.7) is consistent with the action of the Heisenberg group generated by the grade 1 and 2 components in (G.1). Note that the wave function representation (G.2) of the non-abelian Fourier coefficient can be recovered from (G.7) by a Poisson resummation on $M_2$.

## H  Covariantized Polynomials

In the degeneration limit $O(p,q) \to O(p-1,q-1)$ studied in §4, the monomials $\tilde{P}_{\alpha_1\ldots\alpha_i}^{(l)}(Q)$ with $l \geq 0$ are of degree $i - 2l$, and defined by

$$\begin{aligned}
\sum_{l=0}^{1} \tilde{P}_{\gamma\delta}^{(l)}(Q) &= Q_{L\gamma}Q_{L\delta} - \frac{1}{4\pi}\delta_{\gamma\delta}, \\
\tilde{P}_{\delta}^{(0)}(Q) &= Q_{L\delta}, \\
\tilde{P}^{(0)}(Q) &= 1.
\end{aligned} \qquad \text{(H.1)}$$

In the degeneration limit $O(p,q) \to O(p-2,q-2)$ studied in §5.1, the monomials $\mathcal{P}_{\alpha_1\ldots\alpha_i}^{(l)}$ with $l \geq 0$ are of degree $i - 2l$, and defined by

$$\begin{aligned}
\sum_{l=0}^{1} \mathcal{P}_{\gamma\delta}^{(l)}(\Gamma_i,S) &= \Gamma_{L,\gamma\tau}\Gamma_{L,\delta}{}^{\tau} - \frac{1}{4\pi}\delta_{\gamma\delta}, \\
\mathcal{P}_{\delta\tau}^{(0)}(\Gamma) &= Q_{L\delta\tau}, \\
\mathcal{P}^{(0)}(\Gamma) &= 1.
\end{aligned} \qquad \text{(H.2)}$$

For the Abelian rank-2 orbits (5.33), the polynomial are contracted with their matrix-variate Bessel function as

$$\sum_{l=0}^{2} \mathcal{P}_{\alpha\beta,\gamma\delta}^{(l)\mu\nu}(\Gamma_i,S)\widetilde{B}_{\frac{q-5-l}{2}\,\mu\nu}^{(l\,\mathrm{mod}\,2)}(Z) = \delta_{\langle\lambda\kappa,}\delta_{\tau\epsilon\rangle}\,\Gamma_{L\alpha}{}^{\lambda}\Gamma_{L\beta}{}^{\kappa}\Gamma_{L\gamma}{}^{\tau}\Gamma_{L\delta}{}^{\epsilon}\,\delta^{\mu\nu}\widetilde{B}_{\frac{q-5}{2}\,\mu\nu}^{(0)}(Z)$$

$$-\frac{3}{4\pi}\delta_{\langle\alpha\beta,}(\Gamma_{L\gamma}{}^{\kappa}\Gamma_{L\delta\rangle}{}^{\lambda})\widetilde{B}_{\frac{q-6}{2}\,\kappa\lambda}^{(1)}(Z)$$

$$+\frac{3}{16\pi^2}\delta_{\langle\alpha\beta,}\delta_{\gamma\delta\rangle}\delta^{\mu\nu}\widetilde{B}_{\frac{q-7}{2}\,\mu\nu}^{(0)}(Z),$$

$$\sum_{l=0,1}\mathcal{P}_{\rho\beta,\gamma\delta}^{(l)\mu\nu}(\Gamma_i,S)\widetilde{B}_{\frac{q-6-l}{2}\,\mu\nu}^{(l+1\,\mathrm{mod}\,2)}(Z) = \Gamma_{L\,\langle\beta,\rho}\Gamma_{L\gamma}{}^{\tau}\Gamma_{L\,\delta\rangle}{}^{\epsilon}\widetilde{B}_{\frac{q-6}{2}\,\tau\epsilon}^{(1)}(Z)$$

$$-\frac{3}{4\pi}\delta_{\langle\gamma\delta,}\Gamma_{L\beta\rangle}{}^{\kappa}\widetilde{B}_{\frac{q-7}{2}\,\kappa\rho}^{(0)}(Z), \tag{H.3}$$

$$\sum_{l=0,1}\mathcal{P}_{\rho\sigma,\gamma\delta}^{(l)\mu\nu}(\Gamma_i,S)\widetilde{B}_{\frac{q-7-l}{2}\,\mu\nu}^{(l\,\mathrm{mod}\,2)}(Z) = \Gamma_{L\,\gamma,\rho}\Gamma_{L\,\delta,\sigma}\delta^{\mu\nu}\widetilde{B}_{\frac{q-7}{2}\,\mu\nu}^{(0)}(Z)$$

$$-\frac{1}{4\pi}\delta_{\gamma\delta}\widetilde{B}_{\frac{q-8}{2}\,\rho\sigma}^{(1)}(Z),$$

$$\mathcal{P}_{\rho\sigma,\tau\delta}^{(0)\mu\nu}(\Gamma_i,S)\widetilde{B}_{\frac{q-8}{2}\,\mu\nu}^{(1)}(Z) = \Gamma_{L\,\delta,\langle\tau}\widetilde{B}_{\frac{q-8}{2}\,\rho\sigma\rangle}^{(1)}(Z),$$

$$\mathcal{P}_{\rho\sigma,\tau\upsilon}^{(0)\mu\nu}B_{\frac{q-9}{2}\,\mu\nu}^{(0)}(Z) = \delta_{\langle\rho\sigma,}\delta_{\tau\upsilon\rangle}B_{\frac{q-9}{2}\,\mu}^{(0)}{}^{\mu}(Z).$$

For the singular contribution (D.4), the monomials $\mathcal{P}_{\alpha_1\ldots\alpha_i}^{(l_1,l_2)}$ with $l_1,l_2\geq 0$ are of degree $i-2l_1-2l_2$, and defined by

$$\sum_{l_1=0}^{2}\sum_{l_2=0}^{2}\mathcal{P}_{\alpha\beta,\gamma\delta}^{(l_1,l_2)}(\Gamma_1,\Gamma_2,S) = \delta_{\langle\lambda\kappa,}\delta_{\tau\epsilon\rangle}\Gamma_{1L\alpha}{}^{\lambda}\Gamma_{1L\beta}{}^{\kappa}\Gamma_{2L\gamma}{}^{\tau}\Gamma_{2L\delta}{}^{\epsilon} + \delta_{\langle\lambda\kappa,}\delta_{\tau\epsilon\rangle}\Gamma_{2L\alpha}{}^{\lambda}\Gamma_{2L\beta}{}^{\kappa}\Gamma_{1L\gamma}{}^{\tau}\Gamma_{1L\delta}{}^{\epsilon}$$

$$-\frac{3}{4\pi}\big(\delta_{\langle\alpha\beta,}\Gamma_{1L\gamma}\Gamma_{1L\delta\rangle} + \delta_{\langle\alpha\beta,}\Gamma_{2L\gamma}\Gamma_{2L\delta\rangle}\big) - \frac{3}{8\pi^2}\delta_{\langle\alpha\beta,}\delta_{\gamma\delta\rangle},$$

$$\sum_{l_1=0}^{1}\sum_{l_2=0}^{1}\mathcal{P}_{\rho\beta,\gamma\delta}^{(l_1,l_2)}(\Gamma_1,\Gamma_2,S) = \Gamma_{1L\langle\beta,\rho}\Gamma_{2L\gamma}{}^{\tau}\Gamma_{2L\delta\rangle\tau} + \Gamma_{2L\langle\beta,\rho}\Gamma_{1L\gamma}{}^{\tau}\Gamma_{1L\delta\rangle\tau}$$

$$-\frac{3}{4\pi}\big(\delta_{\langle\gamma\delta,}\Gamma_{1L\beta\rangle} + \delta_{\langle\gamma\delta,}\Gamma_{2L\beta\rangle}\big),$$

$$\sum_{l_1=0}^{1}\sum_{l_2=0}^{1}\mathcal{P}_{\rho\sigma,\gamma\delta}^{(l_1,l_2)}(\Gamma_1,\Gamma_2,S) = \Gamma_{1L\gamma\rho}\Gamma_{1L\delta\sigma} + \Gamma_{2L\gamma\rho}\Gamma_{2L\delta\sigma} - \frac{1}{2\pi}\delta_{\rho\sigma}\delta_{\gamma\delta},$$

$$\mathcal{P}_{\rho\sigma,\tau\delta}^{(0,0)}(\Gamma_1,\Gamma_2,S) = \delta_{\langle\rho\sigma,}(\Gamma_{1L\delta\tau\rangle} + \Gamma_{2L\delta\tau\rangle}),$$

$$\mathcal{P}_{\rho\sigma,\tau\upsilon}^{(0,0)}(\Gamma_1,\Gamma_2,S) = \delta_{\langle\rho\sigma,}\delta_{\tau\upsilon\rangle}, \tag{H.4}$$

where $\Gamma = \Gamma_1 + \Gamma_2$.

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
