# Peer review of "Exact effective interactions and 1/4-BPS dyons in heterotic CHL orbifolds"

_SciPost Physics, doi:SciPost Phys. 7, 028 (2019)_

## Round 1 · Referee Report · Anonymous (Referee 1) · 2019-4-19

Strengths

1- Impressive technical computations. The paper combines techniques from string theory, supersymmetry, modular forms in a highly non-trivial way.

Weaknesses

1-This works contains an enormous amount of work for very small payback in terms of physics. While it is impressive that the authors managed to do this computation, it is not clear what qualitative new physics we learn from it.

Report

In this paper the authors conjecture the exact coefficient of a certain six-derivative coupling in the low energy effective action of three-dimensional string vacua with 16 supercharges and check that this coefficient reproduces perturbative corrections up to two loops and has the right form of non-perturbative corrections. The results are also “decompactified” to four dimensions.

The 155 pages of this paper are packed with technical details, and this work certainly demonstrates high-level technical virtuosity. However, one is left to wonder: “why do we care?” Why is this specific coupling worth of 155 pages of hard work? I don’t think the authors provide any convincing argument for that. The introduction nicely summarizes how they came about to study this question, starting from questions regarding precision counting of BPS black holes. However, the paper contains no conclusions or outlook.
There is no discussion of what we have actually gained from this study, in the bigger scheme of things. Of course, we now have a good guess for this 6-derivative coupling, but what qualitative new physics did we learn that we didn’t already know?

While this paper is technically impressive I believe it is of marginal interest for the community. It should be an editorial decision whether SciPost would like to publish papers of this type.

Requested changes

1 The authors should add Conclusions and outlook.

---

## Round 1 · Referee Report · Anonymous (Referee 2) · 2019-4-29

Report

This paper studies a certain six-derivative term in the effective action of three-dimensional string theory with 16 supercharges. Specifically, the paper deals with heterotic string theory compactified on CHL orbifolds times a circle. (The CHL orbifolds are freely acting orbifolds of a six-torus which preserve 16 supersymmetries.)

The basic idea of the paper was presented in an earlier paper by the same authors for the heterotic string on $T^6 \times S^1$, and the present paper is a (non-trivial) extension of those ideas. The effective coupling under consideration receives contributions from 1/2-BPS and 1/4-BPS objects running in loops, and the conjecture of the paper is that the coupling is given by an integral of a certain Siegel modular form over the Siegel upper half-plane. Many string theorists would be familiar with the simpler 1/2-BPS version of this formula (which governs a different 4-derivative term) in the context of threshold corrections to couplings in four-dimensional string compactifications -- the corresponding formula involves an integral over the fundamental domain of the modular group, and the integrand is a product of a certain meromorphic modular form which is a counting function of BPS states (the elliptic genus of the internal manifold) and a non-holomorphic lattice sum over momenta and windings. Here the structure is formally similar: the integrand is a meromorphic modular form which counts BPS states times a lattice sum.

The details, however, seem to involve conceptual as well as technical leaps. At the conceptual level, while the string threshold corrections were derived from the fundamental string theory, the corresponding integral here is a conjecture made by the authors, and involves a sum over a so-called "non-perturbative Narain lattice" which is roughly-speaking the sum of all 1/4-BPS states in string theory. At a technical level, it is not a priori clear that this formula is well-defined, nor that it suffices, and its success points to a very detailed check of the amazing internal consistency of string theory. Another way of stating this point is that integral under question is over the moduli space of string-like objects (but containing the full non-perturbative spectrum of string theory), and it is not clear why e.g. there is not a higher-dimensional moduli space.

Recommendation: If this conceptual point (why only moduli space of 2d surfaces) has been addressed previously then it would be worth commenting on this in the paper, else it could be highlighted so that it is clear to the reader that there is a non-trivial leap.

The authors do spend some time on defining the integral -- the essential problem as I see it are infra-red divergences, which they regulate in some manner. Having done that they can calculate various limits (weak string-coupling, decompactification to 4d) of the integral from which they recover known results. There are some new predictions in other directions, e.g. for the exact degeneracies for 1/4-BPS black holes in the CHL models which carry non-primitive charges of a type that has not been studied before, which in principle can be checked independently.

Recommendation: the regulator could be highlighted a little more in the main text because it is important. At the very least, Appendix B.2.4 could be referred to in Sections 1 and/or 2. (At the moment the reference first appears in Section 3 as far as I can see.)

Overall it is an interesting paper, it informs an important program in string theory (finding the structure of all the effective action in various vacua), and certainly deserves to be published. The main point is that there is a new (conjectural) formula for a certain six-derivative effective coupling of the string theory. There are also mathematical spin-offs. For example the $R \to \infty$ limit has a mysterious piece $G^{(I\bar{I})}$ required by supersymmetry, but has not been calculated directly by the unfolding technique. Supersymmetry thus gives a constraint for the as-yet-unknown unfolding technique for meromorphic Siegel modular forms.

My main criticism is as follows. The paper is quite long, and in spite of a very good effort by the authors to maintain clarity (without which it would have been very difficult to read), the nature of the paper is that it has become quite dense with technology that are not familiar to most string theorists.

Recommendation: one specific idea to make it more readable is to have a title like "Main results" where the main results are presented. (It is already a bit unusual that the statements of main results are spread between Sections 1 and 2, and as a result they are a little diffused.)
  • validity: -
  • significance: -
  • originality: -
  • clarity: -
  • formatting: -
  • grammar: -

Author:  Boris Pioline  on 2019-06-07  [id 536]

(in reply to Report 2 on 2019-04-29)
Category:
answer to question

We appreciate the careful reading, positive comments and suggestions for improvements. In order to help the reader, we have inserted a list of the main new results in the introducrtion (top of p6 in revised version). We also included several sentences in the opening paragraphs of Sec 2, 3, 4, 5 with pointers to the relevant subsections where a discussion of the physical implications of our computations can be found, and pointers at appropriate places (e.g. below 2.19) to Section B.1.3 and B.2.4 where the regularization prescription is explained. As for the referee's specific technical comment, we included a footnote 1 on page 8 emphasizing that the Siegel fundamental domain should $not$ be thought of as parametrizing the worldsheet of a physical "non-perturbative string", but rather as a mathematical device (inspired by string perturbation theory) to produce the relevant automorphic form. Similarly, the 'non-perturbative Narain lattice' should $not$ be thought of as a lattice of non-perturbative charges. We also took this opportunity to add a few references, as well as a footnote 6 on p16 on the issue of instanton-anti-instanton contributions. We hope that these clarifications will make the arguments in our paper easier to follow.

---

## Round 2 · Referee Report · Anonymous · 2019-8-23

Report

The authors have satisfactorily taken into account the detailed suggestions that I made and I now recommend publication.

---

## Round 2 · Author Response

Following suggestions by the referees and editor, we made a few minor changes to clarify a few points and help the reader navigate through the paper.
We also took this opportunity to add a few references.

---

## Round 2 · List of Changes

In order to help the reader, we have inserted a list of the main new results in the introduction (top of p6). We included several sentences in the opening paragraphs of Sec 2, 3, 4, 5 with pointers to the relevant subsections where a discussion of the physical implications of our computations can be found, and pointers at appropriate places (e.g. below 2.19) to Section B.1.3 and B.2.4 where the regularization prescription is explained. We included a footnote 1 on page 8 emphasizing that the Siegel fundamental domain should not be thought of as parametrizing the worldsheet of a physical "non-perturbative string", but rather as a mathematical device (inspired by string perturbation theory) to produce the relevant automorphic form. Similarly, the 'non-perturbative Narain lattice' should not be thought of as a lattice of non-perturbative charges. We also took this opportunity to add a few references, as well as a footnote 6 on p16 commenting on instanton-anti-instanton contributions.

---

## Editorial Decision

published